# Unveil Benign Overfitting for Transformer in Vision: Training Dynamics, Convergence, and Generalization

**Jiarui Jiang**[*][1], **Wei Huang**[*][2], **Miao Zhang**[†][1], **Taiji Suzuki**[3][2], **Liqiang Nie**[1]

[1]Harbin Institute of Technology, Shenzhen
[2]RIKEN AIP
[3]University of Tokyo

jiaruij@outlook.com, wei.huang.vr@riken.jp, zhangmiao@hit.edu.cn
taiji@mist.i.u-tokyo.ac.jp, nieliqiang@hit.edu.cn

## Abstract

Transformers have demonstrated great power in the recent development of large foundational models. In particular, the Vision Transformer (ViT) has brought revolutionary changes to the field of vision, achieving significant accomplishments on the experimental side. However, their theoretical capabilities, particularly in terms of generalization when trained to overfit training data, are still not fully understood. To address this gap, this work delves deeply into the *benign overfitting* perspective of transformers in vision. To this end, we study the optimization of a Transformer composed of a self-attention layer with softmax followed by a fully connected layer under gradient descent on a certain data distribution model. By developing techniques that address the challenges posed by softmax and the inter-dependent nature of multiple weights in transformer optimization, we successfully characterized the training dynamics and achieved generalization in post-training. Our results establish a sharp condition that can distinguish between the small test error phase and the large test error regime, based on the signal-to-noise ratio in the data model. The theoretical results are further verified by experimental simulation. To the best of our knowledge, this is the first work to characterize benign overfitting for Transformers.

## 1 Introduction

Transformers (Vaswani et al., 2017) have revolutionized numerous fields in artificial intelligence, ranging from natural language processing (Devlin et al., 2018; OpenAI, 2023), computer vision Dosovitskiy et al. (2020); He et al. (2022), graph Yun et al. (2019); Hu et al. (2020) to reinforcement learning Chen et al. (2021a); Janner et al. (2021). In particular, Vision Transformer (ViT) Dosovitskiy et al. (2020) has been developed to advancing the computer vision tasks (Liu et al., 2021; Khan et al., 2022; Guo et al., 2022; Tolstikhin et al., 2021; Lin et al., 2014) compared to Convolutional neural networks. Since then, with the high performance, ViT emerged as a hot area of research and application. Numerous technologies and methods began to surface, including enhancing ViT's data and resource efficiency (Touvron et al., 2021; Hassani et al., 2021; Chen et al., 2021b; Touvron et al., 2022), reducing heavy computation cost Chen et al. (2022); Zhu et al. (2024); Wang et al. (2023).

Recent success brought by ViT has been inspiring an increasing number of works to understand the ViT through empirical and theoretical studies. The empirical investigation works study the robustness (Bhojanapalli et al., 2021; Naseer et al., 2021; Paul and Chen, 2022; Bai et al., 2021); and the role in

---

[*]Equal contribution
[†]Corresponding author

self-supervision (Caron et al., 2021; Chen et al., 2021c; Oquab et al., 2023) and others. On the other hand side, the theoretical works have been conducted to understand the ViT from the perspective of expressivity (Vuckovic et al., 2020; Edelman et al., 2022), optimization (Zhang et al., 2020; Tian et al., 2023a,b), generalization (Li et al., 2023a), and in-context learning (Huang et al., 2023c; Garg et al., 2022; Bai et al., 2024; Ahn et al., 2024).

Despite the insightful understanding provided by the above investigations, it remains unclear how ViTs generalize to unseen data when they are trained to overfit the training data. In particular, the conditions under which ViTs can exhibit *benign overfitting* (Bartlett et al., 2020), where the test error remains small despite overfitting to the training data, are not well understood. Moreover, prior work on the optimization and generalization of ViTs often focuses on simplified settings, such as linear transformers or unrealistic loss functions, due to technical limitations. In this work, we aim to fill this gap through a feature learning theory (Allen-Zhu and Li, 2020; Cao et al., 2022). Based on a data generative model with $M$ tokens composed of signal tokens and noise tokens and a two-layer ViT with softmax attention, we characterize the training dynamics from a random initialization and the generalization ability of the ViT after convergence. We establish a sharp separation in condition to distinct the benign overfitting and harmful overfitting regime for ViTs. Our contributions are summarized as follows:

- We successfully characterize the optimization of the ViT through three different phases in dynamics that exhibit rich and unique behaviors related to self-attention. Building on the convergence results, we further characterize the generalization bounds on unseen datasets.

- Technically, we develop novel methods to handle non-linear attention and the inter-dependent nature of multiple weights in transformer optimization when training from scratch.

- We establish a sharp separation condition in the signal-to-noise ratio in data to distinguish the benign overfitting and harmful overfitting regimes of ViTs. This separation is then verified by experimental simulation.

## 2  Related Work

**Benign Overfiting in Deep Learning**    The most prominent behavior of deep learning is its breaking of bias-variance trade-off in statistical learning theory, i.e., good generalization on unseen data even with overfitting on training data. This line of research starts from Bartlett et al. (2020) who studied benign overfitting in linear regression, learning data generated by a linear model with additive Gaussian noises. It is shown that at large input dimensions (which leads to over-parameterization), the excessive risk of the interpolation can be asymptotically optimal. Hastie et al. (2022); Wu and Xu (2020) studied linear regression when both the dimension and the number of samples scale together with a fixed ratio. Timor et al. (2023) proposed a unified data model for linear predictors and studied conditions under which benign overfitting occurs in different problems. Besides the studies on linear models (Wang et al., 2021; Zou et al., 2021; Zhou and Ge, 2023), several recent works tried to study benign overfitting in neural networks (Frei et al., 2022; Kornowski et al., 2024; Chatterji et al., 2022; Xu et al., 2023). In particular, Li et al. (2021) investigated the benign overfitting in two-layer neural networks with the first layer parameters fixed at random initialization. Furthermore, *benign overfitting* was characterized in convolution neural networks (Cao et al., 2022; Kou et al., 2023), OOD (Chen et al., 2024), federated learning (Huang et al., 2023b), graph neural network (Huang et al., 2023a) by feature learning theory (Allen-Zhu and Li, 2020; Cao et al., 2022). Unlike existing research on benign overfitting, this work focuses on ViTs.

**Optimization of Transformers**    Towards understanding the optimization of Transformers, Zhang et al. (2020) provided the analysis for adaptive gradient methods. Jelassi et al. (2022) proposed a spatially structured dataset and a simplified ViT model and showed that ViT implicitly learns the spatial structure of the dataset while generalizing. Besides, Li et al. (2023b) provided fine-grained mechanistic understanding of how transformers learn "semantic structure" for BERT-like framework (Devlin et al., 2018). Furthermore, Tian et al. (2023a,b) characterizes the SGD training dynamics of a 1-layer Transformer and multi-layer Transformer respectively. They focused on the unique phenomena and the role of attention in the training dynamics. Huang et al. (2023c) studied the learning dynamics of a one-layer transformer with softmax attention trained via gradient descent in order to in-context learn linear function classes. In addition, Li et al. (2024) provided a theoretical

analysis of the training dynamics of Transformers with nonlinear self-attention within In-Context Learning. In particular, Li et al. (2023a) is most relevant to us, as they also study the training dynamics of ViTs with a similar data model. However, they considered hinge loss and a specific initialization to simplify analysis and did not characterize the harmful overfitting regime. In summary, while the above work studies the optimization dynamics of attention-based models, they do not characterize benign overfitting as we investigate.

# 3   Problem Setup

In this section, we outline the data generation model, the Vision Transformer model, and the gradient descent training algorithm.

**Notations.** For two sequences $\{x_n\}$ and $\{y_n\}$, we denote $x_n = O(y_n)$ if there exist some absolute constant $C > 0$ and $N > 0$ such that $|x_n| \leq C|y_n|$ for all $n \geq N$, denote $x_n = \Omega(y_n)$ if $y_n = O(x_n)$, and denote $x_n = \Theta(y_n)$ if $x_n = O(y_n)$ and $x_n = \Omega(y_n)$. We also denote $x_n = o(y_n)$ if $\lim |x_n/y_n| = 0$. Finally, we use $\widetilde{O}(\cdot)$, $\widetilde{\Omega}(\cdot)$ and $\widetilde{\Theta}(\cdot)$ to hide logarithmic factors in these notations respectively.

**Definition 3.1** (**Data Generation Model**). *Let $\boldsymbol{\mu}_+, \boldsymbol{\mu}_- \in \mathbb{R}^d$ be fixed vectors representing the signals contained in data points, where $\|\boldsymbol{\mu}_+\|_2 = \|\boldsymbol{\mu}_-\|_2 = \|\boldsymbol{\mu}\|_2$ and $\langle \boldsymbol{\mu}_+, \boldsymbol{\mu}_- \rangle = 0$. Then each data point $(\boldsymbol{X}, y)$ with $\boldsymbol{X} = (\boldsymbol{x}_1, \boldsymbol{x}_2, \dots, \boldsymbol{x}_M)^\top \in \mathbb{R}^{M \times d}$ and $y \in \{-1, 1\}$ is generated from the following distribution $D$:*

1. *The label $y$ is generated as a Rademacher random variable.*

2. *If $y = 1$ then $\boldsymbol{x}_1$ is given as $\boldsymbol{\mu}_+$, if $y = -1$ then $\boldsymbol{x}_1$ is given as $\boldsymbol{\mu}_-$, which represents signals.*

3. *A noise vector $\boldsymbol{\xi}_2$ is generated from the Gaussian distribution $\mathcal{N}(0, \tilde{\sigma}_p^2 \cdot (\boldsymbol{I} - \boldsymbol{\mu}_+ \boldsymbol{\mu}_+^\top \cdot \|\boldsymbol{\mu}\|_2^{-2} - \boldsymbol{\mu}_- \boldsymbol{\mu}_-^\top \cdot \|\boldsymbol{\mu}\|_2^{-2}))$.*

4. *Noise vectors $\boldsymbol{\xi}_3, \dots, \boldsymbol{\xi}_M$ is generated i.i.d from the Gaussian distribution $\mathcal{N}(0, \sigma_p^2 \cdot (\boldsymbol{I} - \boldsymbol{\mu}_+ \boldsymbol{\mu}_+^\top \cdot \|\boldsymbol{\mu}\|_2^{-2} - \boldsymbol{\mu}_- \boldsymbol{\mu}_-^\top \cdot \|\boldsymbol{\mu}\|_2^{-2}))$.*

5. *$\boldsymbol{x}_2, \dots, \boldsymbol{x}_M$ are given by $\boldsymbol{\xi}_2, \dots, \boldsymbol{\xi}_M$, which represent noises.*

Our data model is envisioned by considering $M$ tokens of the data points: $\boldsymbol{x}_1, \boldsymbol{x}_2, \dots, \boldsymbol{x}_M$, which can be seen as patches derived from image data. $\boldsymbol{x}_1$ embodies the signal that is inherently linked to the data's class label, while $\boldsymbol{x}_2, \boldsymbol{x}_3, \dots, \boldsymbol{x}_M$ represents the noise, which is not associated with the label. For simplicity, we assume that the noise patches is independently drawn from Gaussian distribution $\mathcal{N}(0, \tilde{\sigma}_p^2 \cdot (\boldsymbol{I} - \boldsymbol{\mu}_+ \boldsymbol{\mu}_+^\top \cdot \|\boldsymbol{\mu}\|_2^{-2} - \boldsymbol{\mu}_- \boldsymbol{\mu}_-^\top \cdot \|\boldsymbol{\mu}\|_2^{-2}))$ and $\mathcal{N}(0, \sigma_p^2 \cdot (\boldsymbol{I} - \boldsymbol{\mu}_+ \boldsymbol{\mu}_+^\top \cdot \|\boldsymbol{\mu}\|_2^{-2} - \boldsymbol{\mu}_- \boldsymbol{\mu}_-^\top \cdot \|\boldsymbol{\mu}\|_2^{-2}))$, ensuring the noise vectors remains orthogonal to the signal vectors $\boldsymbol{\mu}_+$ and $\boldsymbol{\mu}_-$. Also, if $\tilde{\sigma}_p$ is sufficiently larger than $\sigma_p$, the input tokens will become sparse, which is consistent with the empirical observation that the *attention* and activation values in Transformer-based models are usually sparse (Child et al., 2019; Robinson et al., 2023). We denote $\text{SNR} = \|\boldsymbol{\mu}\|_2/(\sigma_p \sqrt{d})$ to represent the signal-to-noise ratio.

**Two-layer Transformer.** We consider a two layer Transformer network with a self-attention layer and a fixed linear layer, which is defined as:

$$f(\boldsymbol{X}, \theta) = \frac{1}{M} \sum_{l=1}^{M} \boldsymbol{\varphi}(\boldsymbol{x}_l^\top \boldsymbol{W}_Q \boldsymbol{W}_K^\top \boldsymbol{X}^\top) \boldsymbol{X} \boldsymbol{W}_V \boldsymbol{w}_O. \tag{1}$$

Here, $\boldsymbol{\varphi}(\cdot) : \mathbb{R}^M \to \mathbb{R}^M$ denote the softmax function, $\boldsymbol{W}_Q, \boldsymbol{W}_K \in \mathbb{R}^{d \times d_h}, \boldsymbol{W}_V \in \mathbb{R}^{d \times d_v}$ denote the query matrix, key matrix, and value matrix, respectively, and $\boldsymbol{w}_O \in \mathbb{R}^{d_v}$ denote the weight for the linear layer. We use $\theta$ to denote the collection of all the model weights. This model is a simplified version of the ViT model from Dosovitskiy et al. (2020), making our analysis focus on the self-attention mechanism which is the most critical component of the ViTs.

Given a training data set $S = \{(\boldsymbol{X}_n, y_n)\}_{n=1}^N$ generated from the distribution $D$ defined in Definition 3.1, where the subscript $n$ represents the $n$-th sample, we train the two-layer Transformer by

minimizing the empirical cross-entropy loss function:

$$L_S(\theta) = \frac{1}{N} \sum_{n=1}^{N} \ell(y_n f(\boldsymbol{X}_n, \theta)),$$

where $\ell(z) = \log(1 + \exp(-z))$ and $f(\boldsymbol{X}, \theta)$ is the two-layer Transformer. We further defined the population loss (test loss) $L_D(\theta) := \mathbb{E}_{(\boldsymbol{X}, y) \sim D} \ell(y f(\boldsymbol{X}, \theta))$.

**Training algorithm.** We consider Gaussian initialization for the network weights $\boldsymbol{W}_Q, \boldsymbol{W}_K$ and $\boldsymbol{W}_V$, where each entry of $\boldsymbol{W}_Q$ and $\boldsymbol{W}_K$ is sampled from a Gaussian distribution $\mathcal{N}(0, \sigma_h^2)$, and $\boldsymbol{W}_V$ is sampled from $\mathcal{N}(0, \sigma_V^2)$ at initialization. We use gradient descent to optimize training loss $L_S(\theta)$, and the update of $\boldsymbol{W}_Q, \boldsymbol{W}_K$ and $\boldsymbol{W}_V$ can be written as follows:

$$\boldsymbol{W}^{(t+1)} = \boldsymbol{W}^{(t)} - \eta \cdot \nabla_{\boldsymbol{W}} L_S(\theta(t)),$$

where $\boldsymbol{W}^{(t)}$ can be designated as $\boldsymbol{W}_Q, \boldsymbol{W}_K$ or $\boldsymbol{W}_V$.

Recalling the ViT defined in Eq. (1), we can intuitively analyze its training dynamics and generalization: if more attention is paid to signals, i.e., $\boldsymbol{x}_l^\top \boldsymbol{W}_Q \boldsymbol{W}_K^\top \boldsymbol{\mu}_\pm \geq \boldsymbol{x}_l^\top \boldsymbol{W}_Q \boldsymbol{W}_K^\top \boldsymbol{\xi}_i$ for $i \in [M]/\{1\}$, then the vector $\boldsymbol{\varphi}(\boldsymbol{x}_l^\top \boldsymbol{W}_Q \boldsymbol{W}_K^\top \boldsymbol{X}^\top) \boldsymbol{X}$ has a higher similarity with signals $\boldsymbol{\mu}_\pm$ rather than with noises $\boldsymbol{\xi}_i$. In turn, $\boldsymbol{W}_V$ together with $\boldsymbol{w}_O$ have more chance to learn signals $\boldsymbol{\mu}_\pm$ and utilize them to make prediction. At the same time, if vector $\boldsymbol{W}_V \boldsymbol{w}_O$ learns signals $\boldsymbol{\mu}_\pm$ more than noises $\boldsymbol{\xi}_i$, i.e., $\boldsymbol{\mu}_\pm \boldsymbol{W}_V \boldsymbol{w}_O \geq \boldsymbol{\xi}_i \boldsymbol{W}_V \boldsymbol{w}_O$, then during the gradient descent training, the gradient send to $\boldsymbol{x}_l^\top \boldsymbol{W}_Q \boldsymbol{W}_K^\top \boldsymbol{\mu}_\pm$ will be larger than that send to $\boldsymbol{x}_l^\top \boldsymbol{W}_Q \boldsymbol{W}_K^\top \boldsymbol{\xi}_i$. As a result, more and more attention is paid on signals and the similarity between signals $\boldsymbol{\mu}_\pm$ and $\boldsymbol{W}_V \boldsymbol{w}_O$ becomes increasingly high. Therefore, the Transformer can perform well on new data points. On the contrary, if much attention is paid to some noises in the training dataset and the model utilizes them for making prediction and optimizing the training loss, then the model can fit the training dataset well but might not perform well on the new test data.

# 4 Main Results

This section presents our main theoretical results which characterize the convergence and generalization of the ViT model under different sample size $N$ and signal-to-noise ratio $\text{SNR} = \|\boldsymbol{\mu}\|_2 / (\sigma_p \sqrt{d})$. The results are based on the following conditions.

**Condition 4.1.** *Given a sufficiently small failure probability $\delta > 0$ and a target training loss $\epsilon > 0$, suppose that:(1) Dimension $d_h = \widetilde{\Omega}^1\Big(\max\{\text{SNR}^4, \text{SNR}^{-4}\} N^2 \epsilon^{-2}\Big)$. (2) Dimension $d = \widetilde{\Omega}\Big(\epsilon^{-2} N^2 d_h\Big)$. (3) Training sample size $N = \Omega(\text{polylog}(d))$.(4) The number of input tokens $M = \Theta(1)$. (5) The $\ell_2$-norm of linear layer weights $\|\boldsymbol{w}_O\|_2 = \Theta(1)$. (6) The learning rate $\eta \leq \widetilde{O}(\min\{\|\boldsymbol{\mu}\|_2^{-2}, (\sigma_p^2 d)^{-1}\} \cdot d_h^{-\frac{1}{2}})$. (7) The standard deviation of Gaussian initialization $\sigma_V$ satisfies: $\sigma_V \leq \widetilde{O}\big(\|\boldsymbol{w}_O\|_2^{-1} \cdot \min\{\|\boldsymbol{\mu}\|_2^{-1}, (\sigma_p \sqrt{d})^{-1}\} \cdot d_h^{-\frac{1}{4}}\big)$. (8) The variance of Gaussian initialization $\sigma_h^2$ satisfies: $\min\{\|\boldsymbol{\mu}\|_2^{-2}, (\sigma_p^2 d)^{-1}\} \cdot d_h^{-\frac{1}{2}} \cdot \big(\log(6N^2 M^2/\delta)\big)^{-2} \leq \sigma_h^2 \leq \min\{\|\boldsymbol{\mu}\|_2^{-2}, (\sigma_p^2 d)^{-1}\} \cdot d_h^{-\frac{1}{2}} \cdot \big(\log(6N^2 M^2/\delta)\big)^{-\frac{3}{2}}$. (9) Target training loss $\epsilon \leq O(1/\text{polylog}(d))$. (10) The relationship between $\sigma_p$ and $\tilde{\sigma}_p$ satisfies $\tilde{\sigma}_p = C_p \sigma_p$ and $C_p = 5\sqrt{M}$.*

Conditions (1) and (2) ensure an over-parametrized learning setting, and similar conditions have been made in the theoretical analysis of CNN models (Cao et al., 2022; Kou et al., 2023). Condition (3) ensures that there are enough samples in each class with high probability. Conditions (4)-(5) are intended to simplify the calculation, and can be easily generalized to $M = \Omega(1)$, $\|\boldsymbol{w}_O\|_2 = o(1)$ or $\|\boldsymbol{w}_O\|_2 = \omega(1)$ setting. Conditions (6)-(8) ensure that the Transformer can be effectively trained well. Condition (9) ensures that the Transformer is sufficiently overfitting the training data. Condition (10) ensures the sparsity of the input token features.

---

[1] We use $\widetilde{\Omega}(\cdot)$ and $\widetilde{O}(\cdot)$ to omit logarithmic terms in the notation.

**Theorem 4.1** (**Benign Overfiting**). *Under Condition 4.1, if $N \cdot \mathrm{SNR}^2 = \Omega(1)$, then with probability at least $1 - d^{-1}$, there exist $T = \Theta(\eta^{-1}\epsilon^{-1}\|\boldsymbol{\mu}\|_2^{-2}\|\boldsymbol{w}_O\|_2^{-2})$ such that:*

    *1. The training loss converges to $\epsilon$: $L_S(\theta(T)) \leq \epsilon$.*

    *2. The test loss is nearly zero: $L_D(\theta(T)) \leq o(1)$.*

Theorem 4.1 characterizes the case of *benign overfitting*. It shows that as long as $N \cdot \mathrm{SNR}^2 = \Omega(1)$, the Transformer can generalize well, even though it overfit the training data. To complement the above result and highlight the sharpness of this condition, we present the following theorem for the regime of *harmful overfitting*.

**Theorem 4.2** (**Harmful Overfitting**). *Under Condition 4.1, if $N^{-1} \cdot \mathrm{SNR}^{-2} = \Omega(1)$, then with probability at least $1 - d^{-1}$, there exist $T = \Theta(N\eta^{-1}\epsilon^{-1}\sigma_p^{-2}d^{-1}\|\boldsymbol{w}_O\|_2^{-2})$ such that:*

    *1. The training loss converges to $\epsilon$: $L_S(\theta(T)) \leq \epsilon$.*

    *2. The test loss is high: $L_D(\theta(T)) = \Theta(1)$.*

Theorem 4.2 shows that if $N^{-1} \cdot \mathrm{SNR}^{-2} = \Omega(1)$, the trained Transformer has a high test loss as a result of overfitting the noises in the training data. Theorem 4.1 and Theorem 4.2 reveals a sharp phase transition between *benign overfitting* and *harmful overfitting*.

**Comparison with other results.** Firstly, compared with CNNs with $\mathrm{ReLU}^q$ network result (Cao et al., 2022), when signal-to-noise ratio is small ($\mathrm{SNR} \leq 1$), our results show that ViTs require less number of samples to gengeralize well, which reflects the advantage of Transformers. Secondly, previous theoretical benign overfitting analysis of CNNs with ReLU activations rely heavily on signal strength $\|\boldsymbol{\mu}\|_2$ and signal-to-noise ratio SNR (Kou et al., 2023; Meng et al., 2023), i.e., $\mathrm{SNR}^2 = \widetilde{O}(1/N)$ and require $\|\boldsymbol{\mu}\|_2$ to be large enough, while our analysis does not need to impose restrictions on these conditions. Thirdly, previous works only show when Transformer can generalize well (Li et al., 2023a; Deora et al., 2023), while we demonstrate harmful overfitting as a complementary, which reflect that our benign overfitting condition is tighter and more precise.

## 5 Proof Sketch

This section discusses our main challenges in studying ViT's training dynamic, and presents our technical solutions to overcoming them. The complete proofs are given in the appendix.

### 5.1 Vectorized Q & K and scalarized V

Our first main challenge is to deal with the three matrices $\boldsymbol{W}_Q$, $\boldsymbol{W}_K$ and $\boldsymbol{W}_V$. In contrast to CNN models, whose convolutional kernels can be treated as vectors and thus allow for a more straightforward analytical approach (Cao et al., 2022; Kou et al., 2023), the QKV matrices within the Transformer model are inherently complex to analyze. Moreover, their mutual interactions further complicate the analysis. To circumvent this complexity, Oymak et al. (2023) and Tian et al. (2023a) merge the key-query weights(e.g. $\boldsymbol{W} := \boldsymbol{W}_Q \boldsymbol{W}_K^\top$), Huang et al. (2023c); Zhang et al. (2024); Jelassi et al. (2022) employ a specific initialization (e.g. $\boldsymbol{W}_Q^{(0)} = \boldsymbol{W}_K^{(0)} = \boldsymbol{I}$). Although their approach simplifies the analysis, it is not conducive to showing the dynamics of QKV interactions, i.e., how $\boldsymbol{W}_Q$, $\boldsymbol{W}_K$, and $\boldsymbol{W}_V$ affect each other.

In order to simplify the analysis of QKV dynamics without losing rigor, we propose key techniques called *Vectorized Q & K* and *scalarized V*. The basic idea comes from the property that the product of token feature vectors with the QKV matrices results in vectors, e.g., $\boldsymbol{\mu}_+^\top \boldsymbol{W}_Q$, $\boldsymbol{\xi}_2^\top \boldsymbol{W}_Q$, etc., which are more amenable to analysis. Further, each entry of matrix $\boldsymbol{X}\boldsymbol{W}_Q \boldsymbol{W}_K^\top \boldsymbol{X}^\top$ can be regarded as the inner product of two vectors, e.g., $\boldsymbol{\mu}_+^\top \boldsymbol{W}_Q \boldsymbol{W}_K^\top \boldsymbol{\mu}_+ = \langle \boldsymbol{\mu}_+^\top \boldsymbol{W}_Q, \boldsymbol{\mu}_+^\top \boldsymbol{W}_K \rangle$. Therefore, the dynamics of *attention* can be studied by analyzing the dynamics of the *vectorized Q & K* defined as follows:

**Definition 5.1** (Vectorized Q & K). *Let* $\boldsymbol{W}_Q^{(t)}$ *and* $\boldsymbol{W}_K^{(t)}$ *be the QK matrices of the ViT at the t-th iteration of gradient descent. Then we define the vectorized Q and vectorized K as follows*

$$\boldsymbol{q}_+^{(t)} = \boldsymbol{\mu}_+^\top \boldsymbol{W}_Q^{(t)}, \qquad \boldsymbol{q}_-^{(t)} = \boldsymbol{\mu}_-^\top \boldsymbol{W}_Q^{(t)}, \qquad \boldsymbol{q}_{n,i}^{(t)} = \boldsymbol{\xi}_{n,i}^\top \boldsymbol{W}_Q^{(t)},$$

$$\boldsymbol{k}_+^{(t)} = \boldsymbol{\mu}_+^\top \boldsymbol{W}_K^{(t)}, \qquad \boldsymbol{k}_-^{(t)} = \boldsymbol{\mu}_-^\top \boldsymbol{W}_K^{(t)}, \qquad \boldsymbol{k}_{n,i}^{(t)} = \boldsymbol{\xi}_{n,i}^\top \boldsymbol{W}_K^{(t)},$$

*for* $i \in [M] \backslash \{1\}, n \in [N]$.

With Definition 5.1, denoting $S_+ := \{n \in [N] : y_n = 1\}$, $S_- := \{n \in [N] : y_n = -1\}$, we further analyze the dynamics of the *vectorized Q & K*. By carefully computing the dynamic of $\boldsymbol{q}_+^{(t)}$, we have

$$\boldsymbol{q}_+^{(t+1)} - \boldsymbol{q}_+^{(t)} = \frac{\eta}{NM} \sum_{n \in S_+} -\ell_n'^{(t)} \|\boldsymbol{\mu}\|_2^2 \boldsymbol{w}_O^\top \boldsymbol{W}_V^{(t)\top} \boldsymbol{X}_n^\top (diag(\boldsymbol{\varphi}_{n,1}^{(t)}) - \boldsymbol{\varphi}_{n,1}^{(t)\top} \boldsymbol{\varphi}_{n,1}^{(t)}) \boldsymbol{X}_n \boldsymbol{W}_K^{(t)}, \quad (2)$$

where $\boldsymbol{\varphi}_{n,i}^{(t)} := \varphi(\boldsymbol{x}_{n,i}^\top \boldsymbol{W}_Q^{(t)} \boldsymbol{W}_K^{(t)\top} \boldsymbol{X}_n^\top)$ is a shorthand notation, and $\boldsymbol{w}_O^\top \boldsymbol{W}_V^{(t)\top} \boldsymbol{X}_n^\top$ and $\boldsymbol{X}_n \boldsymbol{W}_K^{(t)}$ can be viewed in the following forms

$$\boldsymbol{w}_O^\top \boldsymbol{W}_V^{(t)\top} \boldsymbol{X}_n^\top = \left(\boldsymbol{\mu}_+^\top \boldsymbol{W}_V^{(t)} \boldsymbol{w}_O, \boldsymbol{\xi}_{n,2}^\top \boldsymbol{W}_V^{(t)} \boldsymbol{w}_O, \dots, \boldsymbol{\xi}_{n,M}^\top \boldsymbol{W}_V^{(t)} \boldsymbol{w}_O\right),$$

$$\boldsymbol{X}_n \boldsymbol{W}_K^{(t)} = \left(\boldsymbol{k}_+^{(t)\top}, \boldsymbol{k}_{n,2}^{(t)\top}, \dots, \boldsymbol{k}_{n,M}^{(t)\top}\right)^\top.$$

Thus $\boldsymbol{q}_+^{(t+1)} - \boldsymbol{q}_+^{(t)}$ can be decomposed into a linear combination of $\boldsymbol{k}_+^{(t)}$ and $\boldsymbol{k}_{n,i}^{(t)}$. Therefore, Eq. (2) can be further expanded as $\boldsymbol{q}_+^{(t+1)} - \boldsymbol{q}_+^{(t)} = \alpha_{+,+}^{(t)} \boldsymbol{k}_+^{(t)} + \sum_{n \in S_+} \sum_{i=2}^M \alpha_{n,+,i}^{(t)} \boldsymbol{k}_{n,i}^{(t)}$, where $(\alpha_{+,+}^{(t)}, \alpha_{n,+,2}^{(t)}, \dots, \alpha_{n,+,M}^{(t)}) = \frac{\eta}{NM} \sum_{n \in S_+} -\ell_n'^{(t)} \|\boldsymbol{\mu}\|_2^2 \boldsymbol{w}_O^\top \boldsymbol{W}_V^{(t)\top} \boldsymbol{X}_n^\top (diag(\boldsymbol{\varphi}_{n,1}^{(t)}) - \boldsymbol{\varphi}_{n,1}^{(t)\top} \boldsymbol{\varphi}_{n,1}^{(t)})$. Intuitively, if $\alpha_{+,+}^{(t)}$ is larger enough than $\alpha_{n,+,i}^{(t)}$, then $\langle \boldsymbol{q}_+^{(t)}, \boldsymbol{k}_+^{(t)} \rangle$ will grow faster than $\langle \boldsymbol{q}_+^{(t)}, \boldsymbol{k}_{n,i}^{(t)} \rangle$, which means token $\boldsymbol{\mu}_+$ will pay more *attention* to $\boldsymbol{\mu}_+$ rather than $\boldsymbol{\xi}_{n,i}$. The dynamics of $\boldsymbol{q}_-^{(t)}, \boldsymbol{q}_{n,i}^{(t)}$, $\boldsymbol{k}_+^{(t)}, \boldsymbol{k}_-^{(t)}$ and $\boldsymbol{k}_{n,i}^{(t)}$ are similar to $\boldsymbol{q}_+^{(t)}$, and we can study the dynamics of $QK$ matries by analysing the linear combination coefficients such as $\alpha_{+,+}^{(t)}, \alpha_{n,+,i}^{(t)}$.

As $\boldsymbol{x}_{n,i}^\top \boldsymbol{W}_V^{(t)} \boldsymbol{w}_O$ are scalars and their dynamics contain a factor $\|\boldsymbol{w}_O\|_2^2$, we define the *scalarized V* as follows.

**Definition 5.2** (Scalarized V). *Let* $\boldsymbol{W}_V^{(t)}$ *be the V matrix of the ViT at the t-th iteration of gradient descent. Then there exist coefficients* $\gamma_{V,+}^{(t)}, \gamma_{V,-}^{(t)}, \rho_{V,n,i}^{(t)}$ *such that*

$$\boldsymbol{\mu}_+^\top \boldsymbol{W}_V^{(t)} \boldsymbol{w}_O = \boldsymbol{\mu}_+^\top \boldsymbol{W}_V^{(0)} \boldsymbol{w}_O + \gamma_{V,+}^{(t)} \|\boldsymbol{w}_O\|_2^2,$$

$$\boldsymbol{\mu}_-^\top \boldsymbol{W}_V^{(t)} \boldsymbol{w}_O = \boldsymbol{\mu}_-^\top \boldsymbol{W}_V^{(0)} \boldsymbol{w}_O + \gamma_{V,-}^{(t)} \|\boldsymbol{w}_O\|_2^2,$$

$$\boldsymbol{\xi}_{n,i}^\top \boldsymbol{W}_V^{(t)} \boldsymbol{w}_O = \boldsymbol{\xi}_{n,i}^\top \boldsymbol{W}_V^{(0)} \boldsymbol{w}_O + \rho_{V,n,i}^{(t)} \|\boldsymbol{w}_O\|_2^2$$

*for* $i \in [M] \backslash \{1\}, n \in [N]$.

With the *Vectorized Q & K* and *scalarized V*, one can simplify the study of the Transformer learning process to a meticulous calculation of the coefficients such as $\alpha, \gamma, \rho$ throughout the training period. So how to calculate the dynamics of these coefficients is a key point in our analysis. In the next subsection we describe how to handle *softmax* function and further give bounds for these coefficients.

## 5.2 Dealing with the softmax function

Our second challenge is to deal with the *softmax* function, which is the critical component and introduces non-linear transformation in our Transformer model.

As $\boldsymbol{W}_Q$ and $\boldsymbol{W}_K$ are within the *softmax* function and $\boldsymbol{W}_V$ is outside the *softmax* function, we divide the dynamic of training process into two key aspects: (1) How $\boldsymbol{W}_Q$ and $\boldsymbol{W}_K$ affect $\boldsymbol{W}_V$; (2) How $\boldsymbol{W}_V$ affects $\boldsymbol{W}_Q$ and $\boldsymbol{W}_K$. Next, we present our approach to addressing these two critical issues.

**1. How $W_Q$ and $W_K$ affect $W_V$:** Without loss of generality, we take a data point $(X_n, y_n)$ with $y_n = 1$ as an example and provide the dynamics for $\gamma_{V,+}^{(t)}$ and $\rho_{V,n,i}^{(t)}$ in $W_V$ as follows:

**Lemma 5.1** (Dynamics of $\gamma$ and $\rho$).

$$\gamma_{V,+}^{(t+1)} - \gamma_{V,+}^{(t)} = \frac{\eta\|\boldsymbol{\mu}\|_2^2}{NM} \sum_{n\in S_+, s\in[M]} \frac{-\ell_n'^{(t)}\exp(\boldsymbol{x}_{n,s}^\top W_Q^{(t)} W_K^{(t)\top}\boldsymbol{\mu}_+)}{\exp(\boldsymbol{x}_{n,s}^\top W_Q^{(t)} W_K^{(t)\top}\boldsymbol{\mu}_+) + \sum\limits_{k=2}^{M}\exp(\boldsymbol{x}_{n,s}^\top W_Q^{(t)} W_K^{(t)\top}\boldsymbol{x}_{n,k})}$$

$$|\rho_{V,n,i}^{(t+1)} - \rho_{V,n,i}^{(t)}| \leq \left|\frac{2\eta C_p^2\sigma_p^2 d}{NM} \sum_{s\in[M]} \frac{-\ell_n'^{(t)}\exp(\boldsymbol{x}_{n,s}^\top W_Q^{(t)} W_K^{(t)\top}\boldsymbol{\xi}_{n,i})}{\exp(\boldsymbol{x}_{n,s}^\top W_Q^{(t)} W_K^{(t)\top}\boldsymbol{\mu}_+) + \sum\limits_{k=2}^{M}\exp(\boldsymbol{x}_{n,s}^\top W_Q^{(t)} W_K^{(t)\top}\boldsymbol{\xi}_{n,k})}\right|$$

*for $i \in [M]\backslash\{1\}, n \in S_+$, where $\ell_n'^{(t)} := \ell'(y_n f(X_n, \theta(t)))$ is a shorthand notation.*

In benign overfitting regime where $N \cdot \mathrm{SNR}^2 = \Omega(1)$, the factor $\sum\limits_{n\in S_+} \frac{\eta\|\boldsymbol{\mu}\|_2^2}{NM}$ is larger than $\frac{2\eta C_p^2\sigma_p^2 d}{NM}$. Therefore, as long as $\boldsymbol{x}_{n,s}^\top W_Q^{(t)} W_K^{(t)\top}\boldsymbol{\mu}_+$ are not less than $\boldsymbol{x}_{n,s}^\top W_Q^{(t)} W_K^{(t)\top}\boldsymbol{\xi}_{n,i}$, $\gamma_{V,+}^{(t)}$ will grow faster than $\rho_{V,n,i}^{(t)}$. In other words, if $\boldsymbol{q}_\pm^{(t)}$ and $\boldsymbol{q}_{n,i}^{(t)}$ prefer to align with $\boldsymbol{k}_\pm^{(t)}$ rather than $\boldsymbol{k}_{n,i}^{(t)}$, then $W_V$ would prefer to learn signals rather than memorize noises.

**2. How $W_V$ affects $W_Q$ and $W_K$:** Recalling that $(\alpha_{+,+}^{(t)}, \alpha_{n,+,2}^{(t)}, \ldots, \alpha_{n,+,M}^{(t)}) = \frac{\eta}{NM}\sum\limits_{n\in S_+} -\ell_n'^{(t)}\|\boldsymbol{\mu}\|_2^2 \boldsymbol{w}_O^\top W_V^{(t)\top} X_n^\top (diag(\boldsymbol{\varphi}_{n,1}^{(t)}) - \boldsymbol{\varphi}_{n,1}^{(t)\top}\boldsymbol{\varphi}_{n,1}^{(t)})$, where the most complicated part is the matrix $(diag(\boldsymbol{\varphi}_{n,1}^{(t)}) - \boldsymbol{\varphi}_{n,1}^{(t)\top}\boldsymbol{\varphi}_{n,1}^{(t)})$. We observe that the matrix $diag(\boldsymbol{\varphi}_{n,1}^{(t)}) - \boldsymbol{\varphi}_{n,1}^{(t)\top}\boldsymbol{\varphi}_{n,1}^{(t)}$ has two important properties:

1. The diagonal elements are positive, while the elements on the off-diagonal are negative.
2. The sum of each row and column of this matrix is 0.

Based on properties 1 and 2, we can deduce the following conclusion: as long as $\boldsymbol{\mu}_+^\top W_V^{(t)}\boldsymbol{w}_O$ is larger enough than $\boldsymbol{\xi}_{n,i}^\top W_V^{(t)}\boldsymbol{w}_O$, we have $\alpha_{+,+}^{(t)} \geq 0$ and $\alpha_{n,+,i}^{(t)} \leq 0$, implying that $\boldsymbol{q}_+^{(t)}$ prefers to align $\boldsymbol{k}_+^{(t)}$ rather than $\boldsymbol{k}_{n,i}^{(t)}$.

In summary, if more *attention* is paid to signals, $W_V$ learns the signals faster than it memorizes the noises; conversely, if $W_V$ learns the signals significantly more than it memorizes the noises, more *attention* will be paid to signals rather than noises. In other words, $W_Q$, $W_K$ and $W_V$ promote each other.

### 5.3 Three-Stage Decoupling

Our third main challenge is to deal with the complicated relation among the coefficients, e.g., $\gamma$, $\rho$, $\alpha$. Inspired by the two-stage analysis utilized by Cao et al. (2022) to decouple the coefficients in CNN models, we analyze the ViT training process in three stage. In the following, we explain the key steps for proving Theorem 4.1. The proof for Theorem 4.2 is similar and we detail it in the appendix.

**Stage 1:** The analysis in 5.2 shows that $W_Q$, $W_K$ and $W_V$ can promote each other. But this process of mutual reinforcement requires some conditions, e.g. $\boldsymbol{\mu}_+^\top W_V^{(t)}\boldsymbol{w}_O$ is sufficiently larger than $\boldsymbol{\xi}_{n,i}^\top W_V^{(t)}\boldsymbol{w}_O$ which does not necessarily hold for Gaussian initialization, complicating the proof. Stage 1 was introduced to solve this problem, and Lemma 5.2 formally describes this stage:

**Lemma 5.2** (V's Beginning of Learning Signals). *Under the same conditions as Theorem 4.1, there exist $T_1 = \frac{10M(3M+1)N}{\eta d_h^{\frac{1}{4}}(N\|\boldsymbol{\mu}\|_2^2 - 60M^2 C_p^2\sigma_p^2 d)\|\boldsymbol{w}_O\|_2^2}$ such that the first element of vector $X_n W_V^{(t)}\boldsymbol{w}_O$ dominate its other elements, that is, $\boldsymbol{\mu}_+^\top W_V^{(t)}\boldsymbol{w}_O \geq 3M \cdot |\boldsymbol{\xi}_{n,i}^\top W_V^{(t)}\boldsymbol{w}_O|$ for all $n \in S_+$, $i \in [M]\backslash\{1\}$ and $\boldsymbol{\mu}_-^\top W_V^{(t)}\boldsymbol{w}_O \leq -3M \cdot |\boldsymbol{\xi}_{n,i}^\top W_V^{(t)}\boldsymbol{w}_O|$ for all $n \in S_-$, $i \in [M]\backslash\{1\}$.*

With Lemma 5.2, we can guarantee the monotonicity of *attention* on signals for $t \geq T_1$, which allows us to concentrate on analyzing the growth rate of *attention* in the following stages.

**Stage 2:** Note that the output of *softmax* function has an upper bound 1, thus the output of ViT can be bounded by $\max\{|\boldsymbol{\mu}_\pm^\top \boldsymbol{W}_V^{(t)} \boldsymbol{w}_O|, |\boldsymbol{\xi}_{n,i}^\top \boldsymbol{W}_V^{(t)} \boldsymbol{w}_O|\}$. By Lemma 5.1, it can be proved that there exists $T_2 = \Theta\left(\eta^{-1} \|\boldsymbol{\mu}\|_2^{-2} \|\boldsymbol{w}_O\|_2^{-2} \log(6N^2 M^2/\delta)^{-1}\right)$ such that: $\max\{|\boldsymbol{\mu}_\pm^\top \boldsymbol{W}_V^{(t)} \boldsymbol{w}_O|, |\boldsymbol{\xi}_{n,i}^\top \boldsymbol{W}_V^{(t)} \boldsymbol{w}_O|\} = o(1)$, and further get $1/2 - o(1) \leq -\ell_n^{\prime(t)} \leq 1/2 + o(1)$. Therefore, one can simplify the dynamics of the coefficients (e.g., $\gamma, \rho, \alpha$) by plugging the tight bounds of $-\ell_n^{\prime(t)}$. The following lemma provides some bounds for the dynamics of $\boldsymbol{W}_Q$, $\boldsymbol{W}_K$ and $\boldsymbol{W}_V$ in Stage 2.

**Lemma 5.3** (Dynamics of QKV in Stage 2). *Under Condition 4.1, if* $N \cdot \text{SNR}^2 = \Omega(1)$, *then with probability at least* $1 - d^{-1}$, *there exist constant* $C$, $T_1 = \Theta\left(\eta^{-1} d_h^{-\frac{1}{4}} \|\boldsymbol{\mu}\|_2^{-2} \|\boldsymbol{w}_O\|_2^{-2}\right)$ *and* $T_2 = \widetilde{O}\left(\eta^{-1} \|\boldsymbol{\mu}\|_2^{-2} \|\boldsymbol{w}_O\|_2^{-2}\right)$ *such that:*

$$\boldsymbol{\mu}_+^\top \boldsymbol{W}_V^{(t)} \boldsymbol{w}_O \geq \eta C \|\boldsymbol{\mu}\|_2^2 \|\boldsymbol{w}_O\|_2^2 (t - T_1), \quad \boldsymbol{\mu}_+^\top \boldsymbol{W}_V^{(t)} \boldsymbol{w}_O \geq 3M \cdot |\boldsymbol{\xi}_{n,i}^\top \boldsymbol{W}_V^{(T)} \boldsymbol{w}_O|,$$

$$\boldsymbol{\mu}_-^\top \boldsymbol{W}_V^{(t)} \boldsymbol{w}_O \leq -\eta C \|\boldsymbol{\mu}\|_2^2 \|\boldsymbol{w}_O\|_2^2 (t - T_1), \quad \boldsymbol{\mu}_-^\top \boldsymbol{W}_V^{(t)} \boldsymbol{w}_O \leq -3M \cdot |\boldsymbol{\xi}_{n,i}^\top \boldsymbol{W}_V^{(T)} \boldsymbol{w}_O|,$$

$$\langle \boldsymbol{q}_\pm^{(t)}, \boldsymbol{k}_\pm^{(t)} \rangle - \langle \boldsymbol{q}_\pm^{(t)}, \boldsymbol{k}_{n,j}^{(t)} \rangle \geq \log\left(\exp\left(\langle \boldsymbol{q}_\pm^{(T_1)}, \boldsymbol{k}_\pm^{(T_1)} \rangle - \langle \boldsymbol{q}_\pm^{(T_1)}, \boldsymbol{k}_{n,j}^{(T_1)} \rangle\right) + \frac{\eta^2 C \|\boldsymbol{\mu}\|_2^4 \|\boldsymbol{w}_O\|_2^2 d_h^{\frac{1}{2}}}{N\left(\log(6N^2 M^2/\delta)\right)^2} \cdot (t - T_1)(t - T_1 - 1)\right)$$

$$\langle \boldsymbol{q}_{n,i}^{(t)}, \boldsymbol{k}_\pm^{(t)} \rangle - \langle \boldsymbol{q}_{n,i}^{(t)}, \boldsymbol{k}_{n,j}^{(t)} \rangle \geq \log\left(\exp\left(\langle \boldsymbol{q}_{n,i}^{(T_1)}, \boldsymbol{k}_\pm^{(T_1)} \rangle - \langle \boldsymbol{q}_{n,i}^{(T_1)}, \boldsymbol{k}_{n,j}^{(T_1)} \rangle\right) + \frac{\eta^2 C \sigma_p^2 d \|\boldsymbol{\mu}\|_2^2 \|\boldsymbol{w}_O\|_2^2 d_h^{\frac{1}{2}}}{N\left(\log(6N^2 M^2/\delta)\right)^2} \cdot (t - T_1)(t - T_1 - 1)\right)$$

*for* $i, j \in [M] \backslash \{1\}, n \in [N], t \in [T_1, T_2]$.

Lemma 5.3 presents the training dynamics of $\boldsymbol{W}_Q$, $\boldsymbol{W}_K$ and $\boldsymbol{W}_V$ under benign overfitting regime in two aspects: *direction* and *speed*, that is,

- **direction:** $\boldsymbol{W}_V \boldsymbol{w}_O$ prefer to learn the signals rather than memorize the noises; more and more *attention* is paid to signals, while the *attention* on noises is decreasing.
- **speed:** The inner products of $\boldsymbol{\mu}_\pm$ and $\boldsymbol{W}_V \boldsymbol{w}_O$ grow at a linear rate; the inner products of label-related *vectorized Q & K* grow at a logarithmic rate.

**Stage 3:** As the training process going on, the loss begins to converge, and the loss derivatives $-\ell_n^{\prime(t)}$ no longer remain near $1/2$. Therefore, the increasing rate of the inner products of *vectorize Q & K* and $\boldsymbol{\mu}_\pm^\top \boldsymbol{W}_V \boldsymbol{w}_O$ begins to diminish. Based on the analysis in Stage 2, the *attention* is sufficiently sparse at the beginning of Stage 3. In other words, the *attention* on the signals is nearly 1, while the *attention* on the noises is nearly 0. According to this sparsity, $\boldsymbol{\mu}_\pm^\top \boldsymbol{W}_V^{(t)} \boldsymbol{w}_O$ will grow much faster than $\boldsymbol{\xi}_{n,i}^\top \boldsymbol{W}_V^{(t)} \boldsymbol{w}_O$ by Lemma 5.1, and Lemma 5.4 provides the lower bound of $\boldsymbol{\mu}_\pm^\top \boldsymbol{W}_V^{(t)} \boldsymbol{w}_O$.

**Lemma 5.4.** *Under the same conditions as Theorem 4.1, there exists* $T_3 = \Theta\left(\eta^{-1} \epsilon^{-1} \|\boldsymbol{\mu}\|_2^{-2} \|\boldsymbol{w}_O\|_2^{-2} \log(6N^2 M^2/\delta)^{-1}\right)$ *such that:*

$$\boldsymbol{\mu}_+^\top \boldsymbol{W}_V^{(t)} \boldsymbol{w}_O \geq \log\left(\exp(\boldsymbol{\mu}_+^\top \boldsymbol{W}_V^{(T_2)} \boldsymbol{w}_O) + \eta C \|\boldsymbol{\mu}\|_2^2 \|\boldsymbol{w}_O\|_2^2 (t - T_2)\right),$$

$$\boldsymbol{\mu}_-^\top \boldsymbol{W}_V^{(t)} \boldsymbol{w}_O \leq -\log\left(\exp(-\boldsymbol{\mu}_-^\top \boldsymbol{W}_V^{(T_2)} \boldsymbol{w}_O) + \eta C \|\boldsymbol{\mu}\|_2^2 \|\boldsymbol{w}_O\|_2^2 (t - T_2)\right)$$

*for* $t \in [T_2, T_3]$, *where* $C$ *is a constant,* $T_2$ *is the last iteration of stage 2.*

**Convergence:** Lemma 5.4 provides logarithmic lower bounds for $\boldsymbol{\mu}_\pm^\top \boldsymbol{W}_V^{(t)} \boldsymbol{w}_O$. Plugging $t = T_3$ into these inequality, we have $|\boldsymbol{\mu}_\pm^\top \boldsymbol{W}_V^{(t)} \boldsymbol{w}_O| \geq \log\left(\Theta(1/\epsilon)\right)$, then as long as $|\boldsymbol{\xi}_{n,i}^\top \boldsymbol{W}_V^{(t)} \boldsymbol{w}_O|$ is sufficiently small, it can be proved that $y_n f(\boldsymbol{X}_n, \theta(t)) \geq \log\left(1/\epsilon\right)$ for all $n \in [N]$, thus $\ell(y_n f(\boldsymbol{X}_n, \theta(t))) = \log(1 + \exp(-\log(1/\epsilon))) \leq \epsilon$ and $L_S(\theta(t)) \leq \epsilon$. The convergence of ViT is accordingly obtained.

**Generalization:** Consider a new data point $(\boldsymbol{X}, y)$ generated from the distribution defined in Definition 3.1. Without loss of generality, we suppose that the signal token is $\boldsymbol{\mu}_+$ and the label is 1, i.e. $\boldsymbol{X} = (\boldsymbol{\mu}_+, \boldsymbol{\xi}_2, \dots, \boldsymbol{\xi}_M)$, $y = 1$. It is clear that $\boldsymbol{\xi}_i^\top \boldsymbol{W}_Q \boldsymbol{W}_K^\top \boldsymbol{\mu}_\pm$ has a mean zero, which implies that the *attention* on the signals in the test data may not necessarily be as high as that in the training data. However, the sparsity of *attention* during training process creates conditions for $\boldsymbol{W}_V \boldsymbol{w}_O$ to utilize the signals to make predictions, facilitating the generalization of the ViT. We provide the lower bound for the output of ViT on the unseen data with high probability as follows:

**Lemma 5.5.** *Under the same conditions as Theorem 4.1, there exists* $T_3 = \Theta\left(\eta^{-1}\epsilon^{-1}\|\boldsymbol{\mu}\|_2^{-2}\|\boldsymbol{w}_O\|_2^{-2}\log(6N^2M^2/\delta)^{-1}\right)$, *with probability at least* $1 - \delta/N^2M$,

$$yf(\boldsymbol{X}, \theta(T_3)) \geq \frac{\log(C/\epsilon)}{C'} - 1,$$

where $C$ and $C'$ are constants. Lemma 5.5 shows that $yf(\boldsymbol{X}, \theta(T_3))$ is large with high probability, and through careful calculation, we can provide a small bound for test loss. More details are in Appendix D.4.

## 6 Experimental Verification

We present simulation results on synthetic data and MNIST dataset to verify our theoretical results.

**Synthetic data experiments setting:** We follow Definition 3.1 to generate the training set and test set. Specifically, we set token size $M = 16$ and feature dimension $d = 1024$. Without loss of generality, we set $\boldsymbol{\mu}_+ = \|\boldsymbol{\mu}\|_2 \cdot [1, 0, \dots, 0]^\top$ and $\boldsymbol{\mu}_- = \|\boldsymbol{\mu}\|_2 \cdot [0, 1, 0, \dots, 0]^\top$. We generate noise vector $\boldsymbol{\xi}_2$ from the Gaussian distribution $\mathcal{N}(0, \tilde{\sigma}_p^2 \boldsymbol{I})$, where $\tilde{\sigma}_p$ is fixed to 4. Similarly, we generate the other noise vectors $\boldsymbol{\xi}_i$ for $i \in [M]/\{1, 2\}$ from the Gaussian distribution $\mathcal{N}(0, \sigma_p^2 \boldsymbol{I})$ where $\sigma_p$ is fixed to 0.2.

We consider a two-layer Transformer defined in Section 3. The dimensions of matrix $\boldsymbol{W}_Q$, $\boldsymbol{W}_K$ and $\boldsymbol{W}_V$ are set as $d_h = d_v = 512$. The ViT parameters are initialized using PyTorch's default initialization method, and then they are divided by 16 to ensure that the weights are initialized small enough. We train the ViT with full-batch gradient descent and learning rate $\eta = 0.1$, target training loss $\epsilon = 0.01$. We consider different traning sample size $N$ ranging from 2 to 20, and different signal-to noise-ratio SNR ranging from 0.16 to 15.6. We evaluate the test loss with 100 test data points after training loss converges to $\epsilon$. All experiments are performed on an NVIDIA A100 GPU.

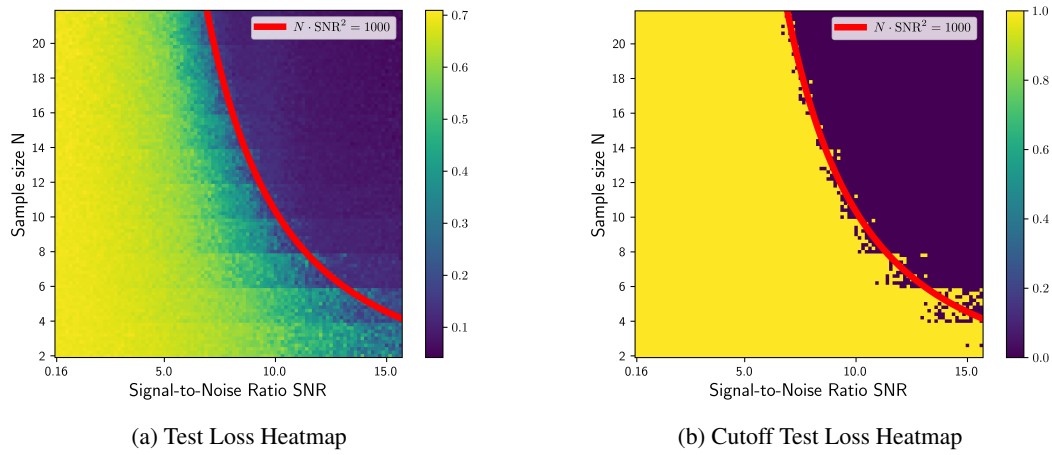

(a) Test Loss Heatmap  (b) Cutoff Test Loss Heatmap

Figure 1: (a) is a heatmap of test loss on synthetic data across various signal-to-noise ratios (SNR) and sample sizes (N). High test losses are indicated with yellow, while low test losses are indicated with purple. (b) is a heatmap that applies a cutoff value 0.2. It categorizes values below 0.2 as 0 (purple), and above 0.2 as 1 (yellow). The expression for the red curves in (a) and (b) is $N \cdot \text{SNR}^2 = 1000$.

**Synthetic data experiments results:** Figure 1a shows that as $N$ and SNR increase, the test loss tends to decrease. As Figure 1b shows, the theoretically derived red curve ($N \cdot \text{SNR}^2 = 1000$)

almost follows the experimental dividing line between the yellow area (test loss > 0.2) and the purple area (test loss < 0.2). These experimental results further validate our theoretical results, that is, the sharp condition separation between benign and harmful overfitting, where $N \cdot \mathrm{SNR}^2 = \Omega(1)$ is a precise condition for benign overfitting and $N^{-1} \cdot \mathrm{SNR}^{-2} = \Omega(1)$ is a precise condition for harmful overfitting. More experimental results on the dynamics of QKV can be found in Appendix A.

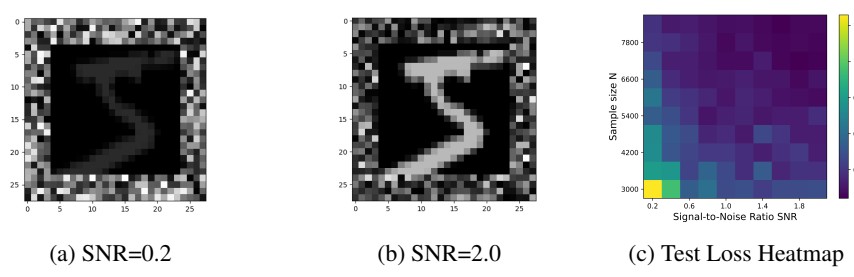

| (a) SNR=0.2 | (b) SNR=2.0 | (c) Test Loss Heatmap |

Figure 2: MNIST experiments

**MNIST experiments setting:** We add Gaussian random noises to the outer regions of the images with a width of 4. The original images and the noises are multiplied by diferent factors to generate diferent dataset with specifc SNR. For example, in Figure 2 (a), we mutiply the original image by 1/6 and the noises by 5/6, thus SNR=0.2; in Figure 2 (b),we mutiply the original image by 2/3 and the noises by 1/3, thus SNR=2.0. We consider a ViT model that consists of two attention layers, each equipped with four self-attention heads, followed by a MLP with ReLU activation. The training sample size N ranges from 3000 to 8400 and the SNR ranges from 0.2 to 2.0.

**MNIST experiments results:** Figure 2 (c) shows the heatmap of test loss, which shows a transition between benign and harmful overfitting regimes. The larger the sample size N and signal-to-noise ratio SNR, the better the generalization performance.

## 7 Conclusion

This paper studies the training dynamics, convergence, and generalization for a two-layer Transformer in vision. By analyzing the *Vectorized Q & K* and *scalarized V* in three-stage decomposition and carefully handling the *softmax* function, we give the precise increasing rate of the *attention* and output of the Transformer in the training process, and further analyze its generalization performance. Our theoretical results reveal a sharp condition separation between benign and harmful overfitting. One limitation of our methodology is our proof technique relies heavily on the sparsity of feature strength, e.g., we assume the standard deviation $\tilde{\sigma}_p$ is sufficiently larger than $\sigma_p$, ensuring that more *attention* is paid to $\boldsymbol{\xi}_2$ rather than other tokens. One future direction is to generalize our analysis to study other abilities of Transformers, such as in-context learning, prompt-tuning and time series forecasting.

## Acknowledgement

We thank the reviewers for their constructive comments on both theoretical and experimental aspects. Miao Zhang was partially sponsored by the National Natural Science Foundation of China under Grant 62306084 and U23B2051, and Shenzhen College Stability Support Plan under Grant GXWD20231128102243003.

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

# Appendix

## Contents

# A More Experimental Results on Training Dynamics

In this section, we follow a similar setting as the experiments in Section 6, and add more experiments. All the experiments in this paper can be performed within hours.

**Experiments setting:** We focus on tracking the dynamics of QKV under benign and harmful overfitting regime, and the key parameters are as follows:

- $M = 4$
- $d = 2048$
- $d_h = 1024$
- $d_v = 256$
- $\sigma_p = 0.05$
- $\tilde{\sigma}_p = 1$

Specifically, we set $n = 16$, $\|\boldsymbol{\mu}\|_2 = 25$ as benign overfitting setting, and set $n = 10$, $\|\boldsymbol{\mu}\|_2 = 15$ as harmful overfitting setting.

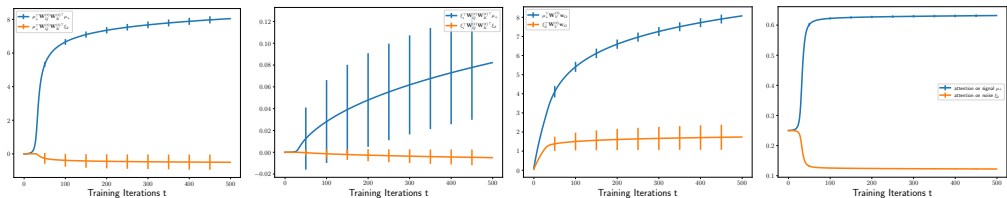

Figure 3: Training Dynamics Under Benign Overfitting Regime

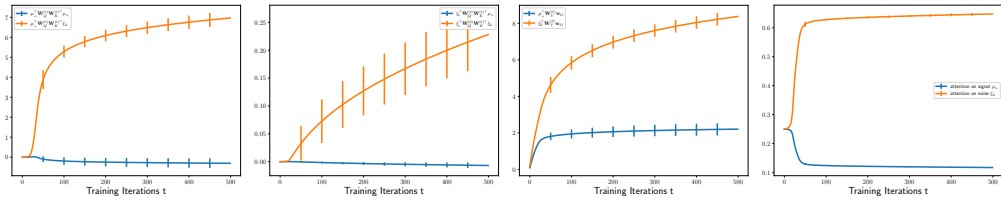

Figure 4: Training Dynamics Under Harmful Overfitting Regime

**Experiments results:** Figure 3 shows the training dynamics under benign overfitting setting, and Figure 4 shows the training dynamics under harmful overfitting setting. In Figure 3, $\boldsymbol{x}_l^\top \boldsymbol{W}_Q \boldsymbol{W}_K^\top \boldsymbol{\mu}_+$ grow faster than $\boldsymbol{x}_l^\top \boldsymbol{W}_Q \boldsymbol{W}_K^\top \boldsymbol{\xi}_2$, and more and more *attention* are paid on signal $\boldsymbol{\mu}_+$. Meanwhile, $\boldsymbol{\mu}_+ \boldsymbol{W}_V \boldsymbol{w}_O$ grows faster than $\boldsymbol{\xi}_2 \boldsymbol{W}_V \boldsymbol{w}_O$. In Figure 4, $\boldsymbol{x}_l^\top \boldsymbol{W}_Q \boldsymbol{W}_K^\top \boldsymbol{\xi}_2$ grow faster than $\boldsymbol{x}_l^\top \boldsymbol{W}_Q \boldsymbol{W}_K^\top \boldsymbol{\mu}_+$, and more and more *attention* are paid on noise $\boldsymbol{\xi}_2$. Meanwhile, $\boldsymbol{\xi}_2 \boldsymbol{W}_V \boldsymbol{w}_O$ grows faster than $\boldsymbol{\mu}_+ \boldsymbol{W}_V \boldsymbol{w}_O$. In other words, the ViT prefers to learn the signals rather than memorize the noises under benign overfitting setting, while it prefers to memorize the noises rather than learn the signals under harmful overfitting setting.

# B Basic Calculation

In this section, we introduce some notations for this paper, and give expressions for the exact gradient of loss function $L_S(\theta)$ with respect to $\boldsymbol{W}_Q, \boldsymbol{W}_K$ and $\boldsymbol{W}_V$.

## B.1 Notations

Table 1: Key notations in this paper

| Symbols | Definitions |
|---|---|
| $\boldsymbol{x}_{n,i}$ | the i-th token in the n-th training sample 
 if $i \in [M]\backslash\{1\}$, $\boldsymbol{x}_{n,i} = \boldsymbol{\xi}_{n,i}$. |
| $\boldsymbol{\varphi}_{n,i}^{(t)}$ | the i-th row of *attention* for the n-th sample, i.e., $\boldsymbol{\varphi}_{n,i}^{(t)} := \boldsymbol{\varphi}(\boldsymbol{x}_{n,i}^{\top}\boldsymbol{W}_Q^{(t)}\boldsymbol{W}_K^{(t)\top}\boldsymbol{X}_n^{\top})$ |
| $S_+, S_-$ | the training samples with +1 labels and -1 labels, 
 i.e. ,$S_+ := \{n \in [N] : y_n = 1\}$, $S_- := \{n \in [N] : y_n = -1\}$ |
| $\boldsymbol{q}_+^{(t)}, \boldsymbol{q}_-^{(t)}, \boldsymbol{q}_{n,i}^{(t)}$ | vectorized Q, defined as $\boldsymbol{q}_+^{(t)} = \boldsymbol{\mu}_+^{\top}\boldsymbol{W}_Q^{(t)}, \boldsymbol{q}_-^{(t)} = \boldsymbol{\mu}_-^{\top}\boldsymbol{W}_Q^{(t)}, \boldsymbol{q}_{n,i}^{(t)} = \boldsymbol{\xi}_{n,i}^{\top}\boldsymbol{W}_Q^{(t)}$ |
| $\boldsymbol{k}_+^{(t)}, \boldsymbol{k}_-^{(t)}, \boldsymbol{k}_{n,i}^{(t)}$ | vectorized K, defined as $\boldsymbol{k}_+^{(t)} = \boldsymbol{\mu}_+^{\top}\boldsymbol{W}_K^{(t)}, \boldsymbol{k}_-^{(t)} = \boldsymbol{\mu}_-^{\top}\boldsymbol{W}_K^{(t)}, \boldsymbol{k}_{n,i}^{(t)} = \boldsymbol{\xi}_{n,i}^{\top}\boldsymbol{W}_K^{(t)}$ |
| $V_+^{(t)}, V_-^{(t)}, V_{n,i}^{(t)}$ | scalarized V, defined as $V_+^{(t)} := \boldsymbol{\mu}_+^{\top}\boldsymbol{W}_V^{(t)}\boldsymbol{w}_O, V_-^{(t)} := \boldsymbol{\mu}_-^{\top}\boldsymbol{W}_V^{(t)}\boldsymbol{w}_O, V_{n,i}^{(t)} := \boldsymbol{\xi}_{n,i}^{\top}\boldsymbol{W}_V^{(t)}\boldsymbol{w}_O$ |
| $\alpha_{\pm,\pm}^{(t)}, \alpha_{n,\pm,i}^{(t)}$ | linear combinations coefficients for the dynamics of $\boldsymbol{q}_+^{(t)}$ and $\boldsymbol{q}_-^{(t)}$, 
 i.e., $\boldsymbol{q}_{\pm}^{(t+1)} - \boldsymbol{q}_{\pm}^{(t)} = \alpha_{\pm,\pm}^{(t)}\boldsymbol{k}_{\pm}^{(t)} + \sum_{n \in S_{\pm}}\sum_{i=2}^{M}\alpha_{n,\pm,i}^{(t)}\boldsymbol{k}_{n,i}^{(t)}$ |
| $\alpha_{n,i,\pm}^{(t)}, \alpha_{n,i,n',i'}^{(t)}$ | linear combinations coefficients for the dynamics of $\boldsymbol{q}_{n,i}^{(t)}$, 
 i.e., $\boldsymbol{q}_{n,i}^{(t+1)} - \boldsymbol{q}_{n,i}^{(t)} = \alpha_{n,i,+}^{(t)}\boldsymbol{k}_+^{(t)} + \alpha_{n,i,-}^{(t)}\boldsymbol{k}_-^{(t)} + \sum_{n'=1}^{N}\sum_{i'=2}^{M}\alpha_{n,i,n',i'}^{(t)}\boldsymbol{k}_{n',i'}^{(t)}$ |
| $\beta_{\pm,\pm}^{(t)}, \beta_{n,\pm,i}^{(t)}$ | linear combinations coefficients for the dynamics of $\boldsymbol{k}_+^{(t)}$ and $\boldsymbol{k}_-^{(t)}$, 
 i.e., $\boldsymbol{k}_{\pm}^{(t+1)} - \boldsymbol{k}_{\pm}^{(t)} = \beta_{\pm,\pm}^{(t)}\boldsymbol{q}_{\pm}^{(t)} + \sum_{n \in S_{\pm}}\sum_{i=2}^{M}\beta_{n,\pm,i}^{(t)}\boldsymbol{q}_{n,i}^{(t)}$ |
| $\beta_{n,i,\pm}^{(t)}, \beta_{n,i,n',i'}^{(t)}$ | linear combinations coefficients for the dynamics of $\boldsymbol{q}_{n,i}^{(t)}$, 
 i.e., $\boldsymbol{k}_{n,i}^{(t+1)} - \boldsymbol{k}_{n,i}^{(t)} = \beta_{n,i,+}^{(t)}\boldsymbol{q}_+^{(t)} + \beta_{n,i,-}^{(t)}\boldsymbol{q}_-^{(t)} + \sum_{n'=1}^{N}\sum_{i'=2}^{M}\beta_{n,i,n',i'}^{(t)}\boldsymbol{q}_{n',i'}^{(t)}$ |
| $softmax(\langle\boldsymbol{q}_{\pm}^{(t)}, \boldsymbol{k}_{\pm}^{(t)}\rangle)$ | a general references to $\dfrac{\exp(\langle\boldsymbol{q}_+^{(t)},\boldsymbol{k}_+^{(t)}\rangle)}{\exp(\langle\boldsymbol{q}_+^{(t)},\boldsymbol{k}_+^{(t)}\rangle)+\sum\limits_{k=2}^{M}\exp(\langle\boldsymbol{q}_+^{(t)},\boldsymbol{k}_{n,k}^{(t)}\rangle)}$ for $n \in S_+$, 
 and $\dfrac{\exp(\langle\boldsymbol{q}_-^{(t)},\boldsymbol{k}_-^{(t)}\rangle)}{\exp(\langle\boldsymbol{q}_-^{(t)},\boldsymbol{k}_-^{(t)}\rangle)+\sum\limits_{k=2}^{M}\exp(\langle\boldsymbol{q}_-^{(t)},\boldsymbol{k}_{n,k}^{(t)}\rangle)}$ for $n \in S_-$ |
| $softmax(\langle\boldsymbol{q}_{n,i}^{(t)}, \boldsymbol{k}_{\pm}^{(t)}\rangle)$ | a general references to $\dfrac{\exp(\langle\boldsymbol{q}_{n,i}^{(t)},\boldsymbol{k}_+^{(t)}\rangle)}{\exp(\langle\boldsymbol{q}_{n,i}^{(t)},\boldsymbol{k}_+^{(t)}\rangle)+\sum\limits_{k=2}^{M}\exp(\langle\boldsymbol{q}_{n,i}^{(t)},\boldsymbol{k}_{n,k}^{(t)}\rangle)}$ for $n \in S_+, i \in [M]\backslash\{1\}$, 
 and $\dfrac{\exp(\langle\boldsymbol{q}_{n,i}^{(t)},\boldsymbol{k}_-^{(t)}\rangle)}{\exp(\langle\boldsymbol{q}_{n,i}^{(t)},\boldsymbol{k}_-^{(t)}\rangle)+\sum\limits_{k=2}^{M}\exp(\langle\boldsymbol{q}_{n,i}^{(t)},\boldsymbol{k}_{n,k}^{(t)}\rangle)}$ for $n \in S_-, i \in [M]\backslash\{1\}$ |
| $softmax(\langle\boldsymbol{q}_{\pm}^{(t)}, \boldsymbol{k}_{n,j}^{(t)}\rangle)$ | a general references to $\dfrac{\exp(\langle\boldsymbol{q}_+^{(t)},\boldsymbol{k}_{n,j}^{(t)}\rangle)}{\exp(\langle\boldsymbol{q}_+^{(t)},\boldsymbol{k}_+^{(t)}\rangle)+\sum\limits_{k=2}^{M}\exp(\langle\boldsymbol{q}_+^{(t)},\boldsymbol{k}_{n,k}^{(t)}\rangle)}$ for $n \in S_+, i \in [M]\backslash\{1\}$, 
 and $\dfrac{\exp(\langle\boldsymbol{q}_-^{(t)},\boldsymbol{k}_{n,j}^{(t)}\rangle)}{\exp(\langle\boldsymbol{q}_-^{(t)},\boldsymbol{k}_-^{(t)}\rangle)+\sum\limits_{k=2}^{M}\exp(\langle\boldsymbol{q}_-^{(t)},\boldsymbol{k}_{n,k}^{(t)}\rangle)}$ for $n \in S_-, i \in [M]\backslash\{1\}$ |
| $softmax(\langle\boldsymbol{q}_{n,i}^{(t)}, \boldsymbol{k}_{n,j}^{(t)}\rangle)$ | a general references to $\dfrac{\exp(\langle\boldsymbol{q}_{n,i}^{(t)},\boldsymbol{k}_{n,j}^{(t)}\rangle)}{\exp(\langle\boldsymbol{q}_{n,i}^{(t)},\boldsymbol{k}_+^{(t)}\rangle)+\sum\limits_{k=2}^{M}\exp(\langle\boldsymbol{q}_{n,i}^{(t)},\boldsymbol{k}_{n,k}^{(t)}\rangle)}$ for $n \in S_+, i,j \in [M]\backslash\{1\}$, 
 and $\dfrac{\exp(\langle\boldsymbol{q}_{n,i}^{(t)},\boldsymbol{k}_{n,j}^{(t)}\rangle)}{\exp(\langle\boldsymbol{q}_{n,i}^{(t)},\boldsymbol{k}_-^{(t)}\rangle)+\sum\limits_{k=2}^{M}\exp(\langle\boldsymbol{q}_{n,i}^{(t)},\boldsymbol{k}_{n,k}^{(t)}\rangle)}$ for $n \in S_-, i,j \in [M]\backslash\{1\}$ |
| $\Lambda_{n,\pm,j}^{(t)}, \Lambda_{n,i,\pm,j}^{(t)}$ | $\Lambda_{n,\pm,j}^{(t)} := \langle\boldsymbol{q}_{\pm}^{(t)},\boldsymbol{k}_{\pm}^{(t)}\rangle - \langle\boldsymbol{q}_{\pm}^{(t)},\boldsymbol{k}_{n,j}^{(t)}\rangle, \quad \Lambda_{n,i,\pm,j}^{(t)} := \langle\boldsymbol{q}_{n,i}^{(t)},\boldsymbol{k}_{\pm}^{(t)}\rangle - \langle\boldsymbol{q}_{n,i}^{(t)},\boldsymbol{k}_{n,j}^{(t)}\rangle$ |
| $\Psi_{\pm}^{(t)}, \Psi_{n,\pm,j}^{(t)}, \Psi_{n,i,\pm}^{(t)}, \Psi_{n,i,j}^{(t)}$ | $\Psi_{\pm}^{(t)} := \langle\boldsymbol{q}_{\pm}^{(t)},\boldsymbol{k}_{n,2}^{(t)}\rangle - \langle\boldsymbol{q}_{\pm}^{(t)},\boldsymbol{k}_{\pm}^{(t)}\rangle, \quad \Psi_{n,\pm,j}^{(t)} := \langle\boldsymbol{q}_{\pm}^{(t)},\boldsymbol{k}_{n,2}^{(t)}\rangle - \langle\boldsymbol{q}_{\pm}^{(t)},\boldsymbol{k}_{n,j}^{(t)}\rangle,$ 
 $\Psi_{n,i,\pm}^{(t)} := \langle\boldsymbol{q}_{n,i}^{(t)},\boldsymbol{k}_{n,2}^{(t)}\rangle - \langle\boldsymbol{q}_{n,i}^{(t)},\boldsymbol{k}_{\pm}^{(t)}\rangle, \quad \Psi_{n,i,j}^{(t)} := \langle\boldsymbol{q}_{n,i}^{(t)},\boldsymbol{k}_{n,2}^{(t)}\rangle - \langle\boldsymbol{q}_{n,i}^{(t)},\boldsymbol{k}_{n,j}^{(t)}\rangle$ |

## B.2 Gradient Calculation

**Gradient of Softmax**. Before computing the gradient of QKV, we solve for the gradient of Softmax function. Suppose there is an input vector $\boldsymbol{a} = (a_1, a_2, \ldots, a_m)$, then the softmax function can be defined as follows:

$$softmax(a_i) = \frac{\exp(a_i)}{\sum\limits_{j=1}^{m} \exp(a_j)}$$

Then we calculate the gradient of $softmax(a_i)$ with respect to $a_i$ and $a_j$ for $i, j \in [m], i \neq j$.

$$
\begin{aligned}
\frac{\partial\, softmax(a_i)}{\partial\, a_i} &= \frac{\exp(a_i) \sum\limits_{j=1}^{m} \exp(a_j) - \exp(a_i)\exp(a_i)}{(\sum\limits_{j=1}^{m} \exp(a_j))^2} \\
&= \frac{\exp(a_i)}{\sum\limits_{j=1}^{m} \exp(a_j)}\left(1 - \frac{\exp(a_i)}{\sum\limits_{j=1}^{m} \exp(a_j)}\right) \\
&= softmax(a_i) \cdot \left(1 - softmax(a_i)\right)
\end{aligned}
\tag{3}
$$

$$
\begin{aligned}
\frac{\partial\, softmax(a_i)}{\partial\, a_j} &= -\frac{\exp(a_i)\exp(a_j)}{(\sum\limits_{j=1}^{m} \exp(a_j))^2} \\
&= softmax(a_i) \cdot softmax(a_j)
\end{aligned}
\tag{4}
$$

(3) and (4) show that the gradient of a Softmax function can be represented as a Jacobi matrix where the elements on the diagonal are $softmax(a_i) \cdot \left(1 - softmax(a_i)\right)$ and the elements on the off-diagonal are $softmax(a_i) \cdot softmax(a_j)$. With these properties, the gradient of our attention vector $\boldsymbol{\varphi}_{n,i}$ ($i \in [M], n \in [N]$) can be expressed as follows

$$\frac{\partial \boldsymbol{\varphi}_{n,i}}{\partial\left(\boldsymbol{x}_{n,i}^{\top}\boldsymbol{W}_Q\boldsymbol{W}_K^{\top}\boldsymbol{X}_n^{\top}\right)} = diag(\boldsymbol{\varphi}_{n,i}) - \boldsymbol{\varphi}_{n,i}^{\top}\boldsymbol{\varphi}_{n,i} \tag{5}$$

**Lemma B.1.** *the gradients of loss function $L_S(\theta)$ with respect to $\boldsymbol{W}_Q, \boldsymbol{W}_K$ and $\boldsymbol{W}_V$ are given by*

$$
\begin{aligned}
\nabla_{\boldsymbol{W}_Q} L_S(\theta) = {}& \frac{1}{NM} \sum_{n \in S_+} \ell_n'(\theta)(\boldsymbol{\mu}_+ \boldsymbol{w}_O^{\top}\boldsymbol{W}_V^{\top}\boldsymbol{X}_n^{\top}(diag(\boldsymbol{\varphi}_{n,1}) - \boldsymbol{\varphi}_{n,1}^{\top}\boldsymbol{\varphi}_{n,1}) \\
&+ \sum_{i=2}^{M} \boldsymbol{\xi}_{n,i}\boldsymbol{w}_O^{\top}\boldsymbol{W}_V^{\top}\boldsymbol{X}_n^{\top}(diag(\boldsymbol{\varphi}_{n,i}) - \boldsymbol{\varphi}_{n,i}^{\top}\boldsymbol{\varphi}_{n,i}))\boldsymbol{X}_n\boldsymbol{W}_K \\
&- \frac{1}{NM} \sum_{n \in S_-} \ell_n'(\theta)(\boldsymbol{\mu}_- \boldsymbol{w}_O^{\top}\boldsymbol{W}_V^{\top}\boldsymbol{X}_n^{\top}(diag(\boldsymbol{\varphi}_{n,1}) - \boldsymbol{\varphi}_{n,1}^{\top}\boldsymbol{\varphi}_{n,1}) \\
&+ \sum_{i=2}^{M} \boldsymbol{\xi}_{n,i}\boldsymbol{w}_O^{\top}\boldsymbol{W}_V^{\top}\boldsymbol{X}_n^{\top}(diag(\boldsymbol{\varphi}_{n,i}) - \boldsymbol{\varphi}_{n,i}^{\top}\boldsymbol{\varphi}_{n,i}))\boldsymbol{X}_n\boldsymbol{W}_K,
\end{aligned}
$$

$$\nabla_{\boldsymbol{W}_K} L_S(\theta) = \frac{1}{NM} \sum_{n \in S_+} \ell_n'(\theta)(\boldsymbol{X}_n^\top (diag(\boldsymbol{\varphi}_{n,1}) - \boldsymbol{\varphi}_{n,1}^\top \boldsymbol{\varphi}_{n,1})\boldsymbol{X}_n \boldsymbol{W}_V \boldsymbol{w}_O \boldsymbol{\mu}_+^\top$$

$$+ \sum_{i=2}^M \boldsymbol{X}_n^\top (diag(\boldsymbol{\varphi}_{n,i}) - \boldsymbol{\varphi}_{n,i}^\top \boldsymbol{\varphi}_{n,i})\boldsymbol{X}_n \boldsymbol{W}_V \boldsymbol{w}_O \boldsymbol{\xi}_{n,i}^\top)\boldsymbol{W}_Q$$

$$- \frac{1}{NM} \sum_{n \in S_-} \ell_n'(\theta)(\boldsymbol{X}_n^\top (diag(\boldsymbol{\varphi}_{n,1}) - \boldsymbol{\varphi}_{n,1}^\top \boldsymbol{\varphi}_{n,1})\boldsymbol{X}_n \boldsymbol{W}_V \boldsymbol{w}_O \boldsymbol{\mu}_-^\top$$

$$+ \sum_{i=2}^M \boldsymbol{X}_n^\top (diag(\boldsymbol{\varphi}_{n,i}) - \boldsymbol{\varphi}_{n,i}^\top \boldsymbol{\varphi}_{n,i})\boldsymbol{X}_n \boldsymbol{W}_V \boldsymbol{w}_O \boldsymbol{\xi}_{n,i}^\top)\boldsymbol{W}_Q,$$

$$\nabla_{\boldsymbol{W}_V} L_S(\theta) = \frac{1}{NM} \sum_{n=1}^N y_n \ell_n'(\theta)[\boldsymbol{w}_O \sum_{l=1}^M \boldsymbol{\varphi}(\boldsymbol{x}_{n,l} \boldsymbol{W}_Q \boldsymbol{W}_K^\top \boldsymbol{X}_n^\top)\boldsymbol{X}_n]^\top$$

*Proof of Lemma B.1.* Considering that in the vector $\boldsymbol{x}_{n,i}^\top \boldsymbol{W}_Q \boldsymbol{W}_K^\top \boldsymbol{X}_n^\top$, the $j$-th element of $\boldsymbol{x}_{n,i}^\top$, i.e. $(\boldsymbol{x}_{n,i})_{[j]}$, is only used to multiply $\boldsymbol{W}_{Q[j,:]}$, then we have

$$\frac{\partial(\boldsymbol{x}_{n,i}^\top \boldsymbol{W}_Q \boldsymbol{W}_K^\top \boldsymbol{X}_n^\top)}{\partial \boldsymbol{W}_{Q[j,:]}} = (\boldsymbol{x}_{n,i})_{[j]} \cdot (\boldsymbol{X}_n \boldsymbol{W}_K)$$

And then we get

$$\nabla_{\boldsymbol{W}_{Q[j,:]}} \big( \boldsymbol{\varphi}(\boldsymbol{x}_{n,i}^\top \boldsymbol{W}_Q \boldsymbol{W}_K^\top \boldsymbol{X}_n^\top)\boldsymbol{X}_n \boldsymbol{W}_V \boldsymbol{w}_O \big)$$
$$= (\boldsymbol{x}_{n,i})_{[j]} \Big[ \frac{\partial(\boldsymbol{\varphi}_{n,i} \boldsymbol{X}_n \boldsymbol{W}_V \boldsymbol{w}_O)}{\partial(\boldsymbol{x}_{n,i}^\top \boldsymbol{W}_Q \boldsymbol{W}_K^\top \boldsymbol{X}_n^\top)} \Big](\boldsymbol{X}_n \boldsymbol{W}_K)$$

Considering all $j \in [d]$, we get

$$\nabla_{\boldsymbol{W}_Q} \big( \boldsymbol{\varphi}(\boldsymbol{x}_{n,i}^\top \boldsymbol{W}_Q \boldsymbol{W}_K^\top \boldsymbol{X}_n^\top)\boldsymbol{X}_n \boldsymbol{W}_V \boldsymbol{w}_O \big)$$
$$= \boldsymbol{x}_{n,i} \Big[ \frac{\partial(\boldsymbol{\varphi}_{n,i} \boldsymbol{X}_n \boldsymbol{W}_V \boldsymbol{w}_O)}{\partial(\boldsymbol{x}_{n,i}^\top \boldsymbol{W}_Q \boldsymbol{W}_K^\top \boldsymbol{X}_n^\top)} \Big] \boldsymbol{X}_n \boldsymbol{W}_K \tag{6}$$

$$\nabla_{\boldsymbol{W}_Q} L_S(\theta) = \frac{1}{N} \sum_{n=1}^{N} y_n \ell'(y_n f(\boldsymbol{X}_n, \theta)) \nabla_{\boldsymbol{W}_Q} f(\boldsymbol{X}_n, \theta)$$

$$= \frac{1}{NM} \sum_{n=1}^{N} y_n \ell'_n(\theta) \sum_{i=1}^{M} \nabla_{\boldsymbol{W}_Q} \left( \boldsymbol{\varphi}(\boldsymbol{x}_{n,i}^\top \boldsymbol{W}_Q \boldsymbol{W}_K^\top \boldsymbol{X}_n^\top) \boldsymbol{X}_n \boldsymbol{W}_V \boldsymbol{w}_O \right)$$

$$= \frac{1}{NM} \sum_{n=1}^{N} y_n \ell'_n(\theta) \sum_{i=1}^{M} \boldsymbol{x}_{n,i} \left[ \frac{\partial (\boldsymbol{\varphi}_{n,i} \boldsymbol{X}_n \boldsymbol{W}_V \boldsymbol{w}_O)}{\partial (\boldsymbol{x}_{n,i}^\top \boldsymbol{W}_Q \boldsymbol{W}_K^\top \boldsymbol{X}_n^\top)} \right] \boldsymbol{X}_n \boldsymbol{W}_K$$

$$= \frac{1}{NM} \sum_{n=1}^{N} y_n \ell'_n(\theta) \sum_{i=1}^{M} \boldsymbol{x}_{n,i} \left[ (\boldsymbol{X}_n \boldsymbol{W}_V \boldsymbol{w}_O)^\top \frac{\partial \boldsymbol{\varphi}_{n,i}}{\partial (\boldsymbol{x}_{n,i}^\top \boldsymbol{W}_Q \boldsymbol{W}_K^\top \boldsymbol{X}_n^\top)} \right] \boldsymbol{X}_n \boldsymbol{W}_K$$

$$= \frac{1}{NM} \sum_{n=1}^{N} y_n \ell'_n(\theta) \sum_{i=1}^{M} \boldsymbol{x}_{n,i} \boldsymbol{w}_O^\top \boldsymbol{W}_V^\top \boldsymbol{X}_n^\top (diag(\boldsymbol{\varphi}_{n,i}) - \boldsymbol{\varphi}_{n,i}^\top \boldsymbol{\varphi}_{n,i}) \boldsymbol{X}_n \boldsymbol{W}_K \qquad (7)$$

$$= \frac{1}{NM} \sum_{n \in S_+} \ell'_n(\theta) (\boldsymbol{\mu}_+ \boldsymbol{w}_O^\top \boldsymbol{W}_V^\top \boldsymbol{X}_n^\top (diag(\boldsymbol{\varphi}_{n,1}) - \boldsymbol{\varphi}_{n,1}^\top \boldsymbol{\varphi}_{n,1})$$

$$+ \sum_{i=2}^{M} \boldsymbol{\xi}_{n,i} \boldsymbol{w}_O^\top \boldsymbol{W}_V^\top \boldsymbol{X}_n^\top (diag(\boldsymbol{\varphi}_{n,i}) - \boldsymbol{\varphi}_{n,i}^\top \boldsymbol{\varphi}_{n,i})) \boldsymbol{X}_n \boldsymbol{W}_K$$

$$- \frac{1}{NM} \sum_{n \in S_-} \ell'_n(\theta) (\boldsymbol{\mu}_- \boldsymbol{w}_O^\top \boldsymbol{W}_V^\top \boldsymbol{X}_n^\top (diag(\boldsymbol{\varphi}_{n,1}) - \boldsymbol{\varphi}_{n,1}^\top \boldsymbol{\varphi}_{n,1})$$

$$+ \sum_{i=2}^{M} \boldsymbol{\xi}_{n,i} \boldsymbol{w}_O^\top \boldsymbol{W}_V^\top \boldsymbol{X}_n^\top (diag(\boldsymbol{\varphi}_{n,i}) - \boldsymbol{\varphi}_{n,i}^\top \boldsymbol{\varphi}_{n,i})) \boldsymbol{X}_n \boldsymbol{W}_K$$

where the third equality is by (6), the fourth equality is by chain rule, the fifth equality is by (5), For the last equality, we expand the equality by materializing all the $\boldsymbol{x}_{n,i}$ (e.g., $\boldsymbol{x}_{n,1} = \boldsymbol{\mu}_+\ for\ n \in S_+$).

Using the similar method, we obtain the gradient of $\boldsymbol{W}_K$.

$$\nabla_{\boldsymbol{W}_K} L_S(\theta) = \frac{1}{N} \sum_{n=1}^{N} y_n \ell'(y_n f(\boldsymbol{X}_n, \theta)) \nabla_{\boldsymbol{W}_K} f(\boldsymbol{X}_n, \theta)$$

$$= \frac{1}{NM} \sum_{n=1}^{N} y_n \ell'_n(\theta) \sum_{i=1}^{M} \nabla_{\boldsymbol{W}_K} \left( \boldsymbol{\varphi}(\boldsymbol{x}_{n,i}^\top \boldsymbol{W}_Q \boldsymbol{W}_K^\top \boldsymbol{X}_n^\top) \boldsymbol{X}_n \boldsymbol{W}_V \boldsymbol{w}_O \right)$$

$$= \frac{1}{NM} \sum_{n=1}^{N} y_n \ell'_n(\theta) \sum_{i=1}^{M} \boldsymbol{X}_n^\top \left[ \frac{\partial (\boldsymbol{\varphi}_{n,i} \boldsymbol{X}_n \boldsymbol{W}_V \boldsymbol{w}_O)}{\partial (\boldsymbol{x}_{n,i}^\top \boldsymbol{W}_Q \boldsymbol{W}_K^\top \boldsymbol{X}_n^\top)} \right]^\top \boldsymbol{x}_{n,i}^\top \boldsymbol{W}_Q$$

$$= \frac{1}{NM} \sum_{n=1}^{N} y_n \ell'_n(\theta) \sum_{i=1}^{M} \boldsymbol{X}_n^\top \left[ \frac{\partial \boldsymbol{\varphi}_{n,i}}{\partial (\boldsymbol{x}_{n,i}^\top \boldsymbol{W}_Q \boldsymbol{W}_K^\top \boldsymbol{X}_n^\top)} (\boldsymbol{X}_n \boldsymbol{W}_V \boldsymbol{w}_O) \right] \boldsymbol{x}_{n,i}^\top \boldsymbol{W}_Q \qquad (8)$$

$$= \frac{1}{NM} \sum_{n \in S_+} \ell'_n(\theta) (\boldsymbol{X}_n^\top (diag(\boldsymbol{\varphi}_{n,1}) - \boldsymbol{\varphi}_{n,1}^\top \boldsymbol{\varphi}_{n,1}) \boldsymbol{X}_n \boldsymbol{W}_V \boldsymbol{w}_O \boldsymbol{\mu}_+^\top$$

$$+ \sum_{i=2}^{M} \boldsymbol{X}_n^\top (diag(\boldsymbol{\varphi}_{n,i}) - \boldsymbol{\varphi}_{n,i}^\top \boldsymbol{\varphi}_{n,i}) \boldsymbol{X}_n \boldsymbol{W}_V \boldsymbol{w}_O \boldsymbol{\xi}_{n,i}^\top) \boldsymbol{W}_Q$$

$$- \frac{1}{NM} \sum_{n \in S_-} \ell'_n(\theta) (\boldsymbol{X}_n^\top (diag(\boldsymbol{\varphi}_{n,1}) - \boldsymbol{\varphi}_{n,1}^\top \boldsymbol{\varphi}_{n,1}) \boldsymbol{X}_n \boldsymbol{W}_V \boldsymbol{w}_O \boldsymbol{\mu}_-^\top$$

$$+ \sum_{i=2}^{M} \boldsymbol{X}_n^\top (diag(\boldsymbol{\varphi}_{n,i}) - \boldsymbol{\varphi}_{n,i}^\top \boldsymbol{\varphi}_{n,i}) \boldsymbol{X}_n \boldsymbol{W}_V \boldsymbol{w}_O \boldsymbol{\xi}_{n,i}^\top) \boldsymbol{W}_Q$$

The gradient of $\boldsymbol{W}_V$ can be obtained using the chain rule as follows

$$\nabla_{\boldsymbol{W}_V} L_S(\theta) = \frac{1}{N} \sum_{n=1}^{N} y_n \ell'(y_n f(\boldsymbol{X}_n, \theta)) \nabla_{\boldsymbol{W}_V} f(\boldsymbol{X}_n, \theta)$$

$$= \frac{1}{NM} \sum_{n=1}^{N} y_n \ell'_n(\theta) [\boldsymbol{w}_O \sum_{l=1}^{M} \boldsymbol{\varphi}(\boldsymbol{x}_{n,l} \boldsymbol{W}_Q \boldsymbol{W}_K^\top (\boldsymbol{X}_n)^\top) \boldsymbol{X}_n]^\top \qquad (9)$$

## B.3 Update Rules

**Definition B.1** (Scalarized V). *Let $\boldsymbol{W}_V^{(t)}$ be the V matrix of the ViT at the t-th iteration of gradient descent. Then there exist coefficients $\gamma_{V,+}^{(t)}$, $\gamma_{V,-}^{(t)}$, $\rho_{V,n,i}^{(t)}$ such that*

$$\boldsymbol{\mu}_+^\top \boldsymbol{W}_V^{(t)} \boldsymbol{w}_O = \boldsymbol{\mu}_+^\top \boldsymbol{W}_V^{(0)} \boldsymbol{w}_O + \gamma_{V,+}^{(t)} \|\boldsymbol{w}_O\|_2^2,$$

$$\boldsymbol{\mu}_-^\top \boldsymbol{W}_V^{(t)} \boldsymbol{w}_O = \boldsymbol{\mu}_-^\top \boldsymbol{W}_V^{(0)} \boldsymbol{w}_O + \gamma_{V,-}^{(t)} \|\boldsymbol{w}_O\|_2^2,$$

$$\boldsymbol{\xi}_{n,i}^\top \boldsymbol{W}_V^{(t)} \boldsymbol{w}_O = \boldsymbol{\xi}_{n,i}^\top \boldsymbol{W}_V^{(0)} \boldsymbol{w}_O + \rho_{V,n,i}^{(t)} \|\boldsymbol{w}_O\|_2^2$$

*for $i \in [M] \backslash \{1\}, n \in [N]$. We further denote $V_+^{(t)} := \boldsymbol{\mu}_+^\top \boldsymbol{W}_V^{(t)} \boldsymbol{w}_O$, $V_-^{(t)} := \boldsymbol{\mu}_-^\top \boldsymbol{W}_V^{(t)} \boldsymbol{w}_O$ and $V_{n,i}^{(t)} := \boldsymbol{\xi}_{n,i}^\top \boldsymbol{W}_V^{(t)} \boldsymbol{w}_O$. We refer to it as scalarized V.*

**Definition B.2** (Vectorized Q & K). *Let $\boldsymbol{W}_Q^{(t)}$ and $\boldsymbol{W}_K^{(t)}$ be the QK matrices of the ViT at the t-th iteration of gradient descent. Then we define the vectorized Q and vectorized K as follows*

$$\boldsymbol{q}_+^{(t)} = \boldsymbol{\mu}_+^\top \boldsymbol{W}_Q^{(t)}, \qquad \boldsymbol{q}_-^{(t)} = \boldsymbol{\mu}_-^\top \boldsymbol{W}_Q^{(t)}, \qquad \boldsymbol{q}_{n,i}^{(t)} = \boldsymbol{\xi}_{n,i}^\top \boldsymbol{W}_Q^{(t)},$$

$$\boldsymbol{k}_+^{(t)} = \boldsymbol{\mu}_+^\top \boldsymbol{W}_K^{(t)}, \qquad \boldsymbol{k}_-^{(t)} = \boldsymbol{\mu}_-^\top \boldsymbol{W}_K^{(t)}, \qquad \boldsymbol{k}_{n,i}^{(t)} = \boldsymbol{\xi}_{n,i}^\top \boldsymbol{W}_K^{(t)}$$

*for $i \in [M] \backslash \{1\}, n \in [N]$.*

**Definition B.3** (Gradient Decomposition). *There exist coefficients $\alpha_{+,+}^{(t)}$, $\alpha_{n,+,i}^{(t)}$, $\alpha_{-,-}^{(t)}$, $\alpha_{n,-,i}^{(t)}$, $\alpha_{n,i,+}^{(t)}$, $\alpha_{n,i,-}^{(t)}$, $\alpha_{n,i,n',i'}^{(t)}$, $\beta_{+,+}^{(t)}$, $\beta_{n,+,i}^{(t)}$, $\beta_{-,-}^{(t)}$, $\beta_{n,-,i}^{(t)}$, $\beta_{n,i,+}^{(t)}$, $\beta_{n,i,-}^{(t)}$, $\beta_{n,i,n',i'}^{(t)}$ such that*

$$\Delta \boldsymbol{q}_+^{(t)} := \boldsymbol{q}_+^{(t+1)} - \boldsymbol{q}_+^{(t)} = \alpha_{+,+}^{(t)} \boldsymbol{k}_+^{(t)} + \sum_{n \in S_+} \sum_{i=2}^{M} \alpha_{n,+,i}^{(t)} \boldsymbol{k}_{n,i}^{(t)},$$

$$\Delta \boldsymbol{q}_-^{(t)} := \boldsymbol{q}_-^{(t+1)} - \boldsymbol{q}_-^{(t)} = \alpha_{-,-}^{(t)} \boldsymbol{k}_-^{(t)} + \sum_{n \in S_-} \sum_{i=2}^{M} \alpha_{n,-,i}^{(t)} \boldsymbol{k}_{n,i}^{(t)},$$

$$\Delta \boldsymbol{q}_{n,i}^{(t)} := \boldsymbol{q}_{n,i}^{(t+1)} - \boldsymbol{q}_{n,i}^{(t)} = \alpha_{n,i,+}^{(t)} \boldsymbol{k}_+^{(t)} + \alpha_{n,i,-}^{(t)} \boldsymbol{k}_-^{(t)} + \sum_{n'=1}^{N} \sum_{i'=2}^{M} \alpha_{n,i,n',i'}^{(t)} \boldsymbol{k}_{n',i'}^{(t)},$$

$$\Delta \boldsymbol{k}_+^{(t)} := \boldsymbol{k}_+^{(t+1)} - \boldsymbol{k}_+^{(t)} = \beta_{+,+}^{(t)} \boldsymbol{q}_+^{(t)} + \sum_{n \in S_+} \sum_{i=2}^{M} \beta_{n,+,i}^{(t)} \boldsymbol{q}_{n,i}^{(t)},$$

$$\Delta \boldsymbol{k}_-^{(t)} := \boldsymbol{k}_-^{(t+1)} - \boldsymbol{k}_-^{(t)} = \beta_{-,-}^{(t)} \boldsymbol{q}_-^{(t)} + \sum_{n \in S_-} \sum_{i=2}^{M} \beta_{n,-,i}^{(t)} \boldsymbol{q}_{n,i}^{(t)},$$

$$\Delta \boldsymbol{k}_{n,i}^{(t)} := \boldsymbol{k}_{n,i}^{(t+1)} - \boldsymbol{k}_{n,i}^{(t)} = \beta_{n,i,+}^{(t)} \boldsymbol{q}_+^{(t)} + \beta_{n,i,-}^{(t)} \boldsymbol{q}_-^{(t)} + \sum_{n'=1}^{N} \sum_{i'=2}^{M} \beta_{n,i,n',i'}^{(t)} \boldsymbol{q}_{n',i'}^{(t)}$$

*for $i, i' \in [M] \backslash \{1\}$ and $n, n' \in [N]$.*

**Remark.** With the scalarized V in B.1, we can portray the learning process of $\boldsymbol{W}_V$ on signals and noises by analyzing the dynamics of $\gamma_{V,+}^{(t)}$, $\gamma_{V,-}^{(t)}$ and $\rho_{V,n,i}^{(t)}$ (or $V_+^{(t)}$, $V_-^{(t)}$ and $V_{n,i}^{(t)}$). Also, as scalars, the effect of $\boldsymbol{W}_V$ on the learning process of $\boldsymbol{W}_Q$ and $\boldsymbol{W}_K$ is more conveniently to describe. With the Vectorized Q & K in B.2, we can decompose $\boldsymbol{x}^\top \boldsymbol{W}_Q \boldsymbol{W}_K \boldsymbol{X}^\top$ into the inner product of vectors $\boldsymbol{q}$ (a generalized reference to $\boldsymbol{q}_+$, $\boldsymbol{q}_-$ and $\boldsymbol{q}_{n,i}$) and $\boldsymbol{k}$ (a generalized reference to $\boldsymbol{k}_+$, $\boldsymbol{k}_-$ and $\boldsymbol{k}_{n,i}$), e.g., $\langle \boldsymbol{q}_+, \boldsymbol{k}_+ \rangle = \boldsymbol{\mu}_+^\top \boldsymbol{W}_Q \boldsymbol{W}_K \boldsymbol{\mu}_+$. Then the dynamics of $\boldsymbol{x}^\top \boldsymbol{W}_Q \boldsymbol{W}_K \boldsymbol{X}^\top$ can be characterized by analyzing the dynamics of the inner products of $\boldsymbol{q}$ and $\boldsymbol{k}$, and then further get the dynamics of attentions. The gradient decomposition in B.3 comes from (7) and (8). Note that (7) ends in $\boldsymbol{X}_n \boldsymbol{W}_K$ and (8) ends in $\boldsymbol{x}_{n,i}^\top \boldsymbol{W}_Q$. This suggests that the dynamics of $\boldsymbol{q}_\pm^{(t)}$ and $\boldsymbol{q}_{n,i}^{(t)}$ can be decomposed into linear combinations of $\boldsymbol{k}_\pm^{(t)}$ and $\boldsymbol{k}_{n,i}^{(t)}$, and similarly, the gradient of $\boldsymbol{k}_\pm^{(t)}$ and $\boldsymbol{k}_{n,i}^{(t)}$ can be decomposed into linear combinations of $\boldsymbol{q}_\pm^{(t)}$ and $\boldsymbol{q}_{n,i}^{(t)}$. We calculate $\alpha$ (a generalized reference to $\alpha_{+,+}$, $\alpha_+^{n,i}$, $\alpha_{-,-}$, etc.) and $\beta$ (a generalized reference to $\beta_{+,+}$, $\beta_{n,+,i}$, $\beta_{-,-}$, etc.) in detail in Appendix F.

**Lemma B.2** (Update Rule for V). *The coefficients $\gamma_{V,+}^{(t)}$, $\gamma_{V,-}^{(t)}$, $\rho_{V,n,i}^{(t)}$ defined in Definition B.1 satisfy the following iterative equations:*

$$
\gamma_{V,+}^{(t+1)} = \gamma_{V,+}^{(t)}
$$
$$
- \frac{\eta \|\boldsymbol{\mu}\|_2^2}{NM} \sum_{n \in S_+} \ell_n'^{(t)} \Big( \frac{\exp(\langle \boldsymbol{q}_+^{(t)}, \boldsymbol{k}_+^{(t)} \rangle)}{\exp(\langle \boldsymbol{q}_+^{(t)}, \boldsymbol{k}_+^{(t)} \rangle) + \sum\limits_{k=2}^{M} \exp(\langle \boldsymbol{q}_+^{(t)}, \boldsymbol{k}_{n,k}^{(t)} \rangle)}
$$
$$
+ \sum_{j=2}^{M} \frac{\exp(\langle \boldsymbol{q}_{n,j}^{(t)}, \boldsymbol{k}_+^{(t)} \rangle)}{\exp(\langle \boldsymbol{q}_{n,j}^{(t)}, \boldsymbol{k}_+^{(t)} \rangle) + \sum\limits_{k=2}^{M} \exp(\langle \boldsymbol{q}_{n,j}^{(t)}, \boldsymbol{k}_{n,k}^{(t)} \rangle)} \Big),
$$

$$
\gamma_{V,-}^{(t+1)} = \gamma_{V,-}^{(t)}
$$
$$
+ \frac{\eta \|\boldsymbol{\mu}\|_2^2}{NM} \sum_{n \in S_-} \ell_n'^{(t)} \Big( \frac{\exp(\langle \boldsymbol{q}_-^{(t)}, \boldsymbol{k}_-^{(t)} \rangle)}{\exp(\langle \boldsymbol{q}_-^{(t)}, \boldsymbol{k}_-^{(t)} \rangle) + \sum\limits_{k=2}^{M} \exp(\langle \boldsymbol{q}_-^{(t)}, \boldsymbol{k}_{n,k}^{(t)} \rangle)}
$$
$$
+ \sum_{j=2}^{M} \frac{\exp(\langle \boldsymbol{q}_{n,j}^{(t)}, \boldsymbol{k}_-^{(t)} \rangle)}{\exp(\langle \boldsymbol{q}_{n,j}^{(t)}, \boldsymbol{k}_-^{(t)} \rangle) + \sum\limits_{k=2}^{M} \exp(\langle \boldsymbol{q}_{n,j}^{(t)}, \boldsymbol{k}_{n,k}^{(t)} \rangle)} \Big),
$$

$$
\rho_{V,n,i}^{(t+1)} = \rho_{V,n,i}^{(t)}
$$
$$
- \frac{\eta}{NM} \sum_{n' \in S_+} \ell_{n'}'^{(t)} \Big( \sum_{i'=2}^{M} (\langle \boldsymbol{\xi}_{n,i}, \boldsymbol{\xi}_{n',i'} \rangle \frac{\exp(\langle \boldsymbol{q}_+^{(t)}, \boldsymbol{k}_{n',i'}^{(t)} \rangle)}{\exp(\langle \boldsymbol{q}_+^{(t)}, \boldsymbol{k}_+^{(t)} \rangle) + \sum\limits_{k=2}^{M} \exp(\langle \boldsymbol{q}_+^{(t)}, \boldsymbol{k}_{n',k}^{(t)} \rangle)}
$$
$$
+ \sum_{j=2}^{M} \langle \boldsymbol{\xi}_{n,i}, \boldsymbol{\xi}_{n',i'} \rangle \frac{\exp(\langle \boldsymbol{q}_{n',j}^{(t)}, \boldsymbol{k}_{n',i'}^{(t)} \rangle)}{\exp(\langle \boldsymbol{q}_{n',j}^{(t)}, \boldsymbol{k}_+^{(t)} \rangle) + \sum\limits_{k=2}^{M} \exp(\langle \boldsymbol{q}_{n',j}^{(t)}, \boldsymbol{k}_{n',k}^{(t)} \rangle)} ))
$$
$$
+ \frac{\eta}{NM} \sum_{n' \in S_-} \ell_{n'}'^{(t)} \Big( \sum_{i'=2}^{M} (\langle \boldsymbol{\xi}_{n,i}, \boldsymbol{\xi}_{n',i'} \rangle \frac{\exp(\langle \boldsymbol{q}_-^{(t)}, \boldsymbol{k}_{n',i'}^{(t)} \rangle)}{\exp(\langle \boldsymbol{q}_-^{(t)}, \boldsymbol{k}_-^{(t)} \rangle) + \sum\limits_{k=2}^{M} \exp(\langle \boldsymbol{q}_-^{(t)}, \boldsymbol{k}_{n',k}^{(t)} \rangle)}
$$
$$
+ \sum_{j=2}^{M} \langle \boldsymbol{\xi}_{n,i}, \boldsymbol{\xi}_{n',i'} \rangle \frac{\exp(\langle \boldsymbol{q}_{n',j}^{(t)}, \boldsymbol{k}_{n',i'}^{(t)} \rangle)}{\exp(\langle \boldsymbol{q}_{n',j}^{(t)}, \boldsymbol{k}_-^{(t)} \rangle) + \sum\limits_{k=2}^{M} \exp(\langle \boldsymbol{q}_{n',j}^{(t)}, \boldsymbol{k}_{n',k}^{(t)} \rangle)} ))
$$

*for $i \in [M] \backslash \{1\}, n \in [N]$.*

*Proof of Lemma B.2.* Base on (9), we have

$$\boldsymbol{x}^\top \nabla_{\boldsymbol{W}_V} L_S(\theta) \boldsymbol{w}_O = \boldsymbol{x}^\top \frac{1}{NM} \sum_{n=1}^N y_n \ell_n'(\theta) [\boldsymbol{w}_O \sum_{l=1}^M \boldsymbol{\varphi}(\boldsymbol{x}_{n,l} \boldsymbol{W}_Q \boldsymbol{W}_K^\top (\boldsymbol{X}_n)^\top) \boldsymbol{X}_n]^\top \boldsymbol{w}_O$$

$$= \frac{1}{NM} \sum_{n=1}^N y_n \ell_n'(\theta) \sum_{l=1}^M \boldsymbol{x}^\top \boldsymbol{X}_n^\top \boldsymbol{\varphi}(\boldsymbol{x}_{n,l} \boldsymbol{W}_Q \boldsymbol{W}_K^\top (\boldsymbol{X}_n)^\top)^\top \|\boldsymbol{w}_O\|_2^2$$

$$= \frac{1}{NM} \sum_{n \in S_+} \ell_n'(\theta) \sum_{l=1}^M \Big( \langle \boldsymbol{x}, \boldsymbol{\mu}_+ \rangle \frac{\exp(\boldsymbol{x}_{n,l}^\top \boldsymbol{W}_Q \boldsymbol{W}_K^\top \boldsymbol{\mu}_+)}{\exp(\boldsymbol{x}_{n,l}^\top \boldsymbol{W}_Q \boldsymbol{W}_K^\top \boldsymbol{\mu}_+) + \sum\limits_{j=2}^M \exp(\boldsymbol{x}_{n,l}^\top \boldsymbol{W}_Q \boldsymbol{W}_K^\top \boldsymbol{\xi}_{n,j})}$$

$$+ \sum_{i=2}^M (\langle \boldsymbol{x}, \boldsymbol{\xi}_{n,i} \rangle \frac{\exp(\boldsymbol{x}_{n,l}^\top \boldsymbol{W}_Q \boldsymbol{W}_K^\top \boldsymbol{\xi}_{n,i})}{\exp(\boldsymbol{x}_{n,l}^\top \boldsymbol{W}_Q \boldsymbol{W}_K^\top \boldsymbol{\mu}_+) + \sum\limits_{j=2}^M \exp(\boldsymbol{x}_{n,l}^\top \boldsymbol{W}_Q \boldsymbol{W}_K^\top \boldsymbol{\xi}_{n,j})})) \|\boldsymbol{w}_O\|_2^2$$

$$- \frac{1}{NM} \sum_{n \in S_-} \ell_n'(\theta) \sum_{l=1}^M \Big( \langle \boldsymbol{x}, \boldsymbol{\mu}_- \rangle \frac{\exp(\boldsymbol{x}_{n,l}^\top \boldsymbol{W}_Q \boldsymbol{W}_K^\top \boldsymbol{\mu}_-)}{\exp(\boldsymbol{x}_{n,l}^\top \boldsymbol{W}_Q \boldsymbol{W}_K^\top \boldsymbol{\mu}_-) + \sum\limits_{j=2}^M \exp(\boldsymbol{x}_{n,l}^\top \boldsymbol{W}_Q \boldsymbol{W}_K^\top \boldsymbol{\xi}_{n,j})}$$

$$+ \sum_{i=2}^M (\langle \boldsymbol{x}, \boldsymbol{\xi}_{n,i} \rangle \frac{\exp(\boldsymbol{x}_{n,l}^\top \boldsymbol{W}_Q \boldsymbol{W}_K^\top \boldsymbol{\xi}_{n,i})}{\exp(\boldsymbol{x}_{n,l}^\top \boldsymbol{W}_Q \boldsymbol{W}_K^\top \boldsymbol{\mu}_-) + \sum\limits_{j=2}^M \exp(\boldsymbol{x}_{n,l}^\top \boldsymbol{W}_Q \boldsymbol{W}_K^\top \boldsymbol{\xi}_{n,j})})) \|\boldsymbol{w}_O\|_2^2$$

$$= \frac{1}{NM} \sum_{n \in S_+} \ell_n'(\theta) \Big( (\langle \boldsymbol{x}, \boldsymbol{\mu}_+ \rangle \frac{\exp(\boldsymbol{\mu}_+^\top \boldsymbol{W}_Q \boldsymbol{W}_K^\top \boldsymbol{\mu}_+)}{\exp(\boldsymbol{\mu}_+^\top \boldsymbol{W}_Q \boldsymbol{W}_K^\top \boldsymbol{\mu}_+) + \sum\limits_{k=2}^M \exp(\boldsymbol{\mu}_+^\top \boldsymbol{W}_Q \boldsymbol{W}_K^\top \boldsymbol{\xi}_{n,k})}$$

$$+ \sum_{j=2}^M \langle \boldsymbol{x}, \boldsymbol{\mu}_+ \rangle \frac{\exp(\boldsymbol{\xi}_{n,j}^\top \boldsymbol{W}_Q \boldsymbol{W}_K^\top \boldsymbol{\mu}_+)}{\exp(\boldsymbol{\xi}_{n,j}^\top \boldsymbol{W}_Q \boldsymbol{W}_K^\top \boldsymbol{\mu}_+) + \sum\limits_{k=2}^M \exp(\boldsymbol{\xi}_{n,j}^\top \boldsymbol{W}_Q \boldsymbol{W}_K^\top \boldsymbol{\xi}_{n,k})})$$

$$+ \sum_{i=2}^M (\langle \boldsymbol{x}, \boldsymbol{\xi}_{n,i} \rangle \frac{\exp(\boldsymbol{\mu}_+ \boldsymbol{W}_Q \boldsymbol{W}_K^\top \boldsymbol{\xi}_{n,i})}{\exp(\boldsymbol{\mu}_+^\top \boldsymbol{W}_Q \boldsymbol{W}_K^\top \boldsymbol{\mu}_+) + \sum\limits_{k=2}^M \exp(\boldsymbol{\mu}_+^\top \boldsymbol{W}_Q \boldsymbol{W}_K^\top \boldsymbol{\xi}_{n,k})}$$

$$+ \sum_{j=2}^M \langle \boldsymbol{x}, \boldsymbol{\xi}_{n,i} \rangle \frac{\exp(\boldsymbol{\xi}_{n,j} \boldsymbol{W}_Q \boldsymbol{W}_K^\top \boldsymbol{\xi}_{n,i})}{\exp(\boldsymbol{\xi}_{n,j}^\top \boldsymbol{W}_Q \boldsymbol{W}_K^\top \boldsymbol{\mu}_+) + \sum\limits_{k=2}^M \exp(\boldsymbol{\xi}_{n,j}^\top \boldsymbol{W}_Q \boldsymbol{W}_K^\top \boldsymbol{\xi}_{n,k})})) \|\boldsymbol{w}_O\|_2^2$$

$$- \frac{1}{NM} \sum_{n \in S_-} \ell_n'(\theta) \Big( (\langle \boldsymbol{x}, \boldsymbol{\mu}_- \rangle \frac{\exp(\boldsymbol{\mu}_-^\top \boldsymbol{W}_Q \boldsymbol{W}_K^\top \boldsymbol{\mu}_-)}{\exp(\boldsymbol{\mu}_-^\top \boldsymbol{W}_Q \boldsymbol{W}_K^\top \boldsymbol{\mu}_-) + \sum\limits_{k=2}^M \exp(\boldsymbol{\mu}_-^\top \boldsymbol{W}_Q \boldsymbol{W}_K^\top \boldsymbol{\xi}_{n,k})}$$

$$+ \sum_{j=2}^M \langle \boldsymbol{x}, \boldsymbol{\mu}_- \rangle \frac{\exp(\boldsymbol{\xi}_{n,j}^\top \boldsymbol{W}_Q \boldsymbol{W}_K^\top \boldsymbol{\mu}_-)}{\exp(\boldsymbol{\xi}_{n,j}^\top \boldsymbol{W}_Q \boldsymbol{W}_K^\top \boldsymbol{\mu}_-) + \sum\limits_{k=2}^M \exp(\boldsymbol{\xi}_{n,j}^\top \boldsymbol{W}_Q \boldsymbol{W}_K^\top \boldsymbol{\xi}_{n,k})})$$

$$+ \sum_{i=2}^M (\langle \boldsymbol{x}, \boldsymbol{\xi}_{n,i} \rangle \frac{\exp(\boldsymbol{\mu}_- \boldsymbol{W}_Q \boldsymbol{W}_K^\top \boldsymbol{\xi}_{n,i})}{\exp(\boldsymbol{\mu}_-^\top \boldsymbol{W}_Q \boldsymbol{W}_K^\top \boldsymbol{\mu}_-) + \sum\limits_{k=2}^M \exp(\boldsymbol{\mu}_-^\top \boldsymbol{W}_Q \boldsymbol{W}_K^\top \boldsymbol{\xi}_{n,k})}$$

$$+ \sum_{j=2}^M \langle \boldsymbol{x}, \boldsymbol{\xi}_{n,i} \rangle \frac{\exp(\boldsymbol{\xi}_{n,j} \boldsymbol{W}_Q \boldsymbol{W}_K^\top \boldsymbol{\xi}_{n,i})}{\exp(\boldsymbol{\xi}_{n,j}^\top \boldsymbol{W}_Q \boldsymbol{W}_K^\top \boldsymbol{\mu}_-) + \sum\limits_{k=2}^M \exp(\boldsymbol{\xi}_{n,j}^\top \boldsymbol{W}_Q \boldsymbol{W}_K^\top \boldsymbol{\xi}_{n,k})})) \|\boldsymbol{w}_O\|_2^2$$

(10)

where the second equality we expand $\boldsymbol{X}_n$ into vectors and make inner products with $x$, the third equality we materializing all the $\boldsymbol{x}_{n,l}$ (e.g., $\boldsymbol{x}_{n,1} = \boldsymbol{\mu}_+$ for $n \in S_+$). Note the orthogonality between $\boldsymbol{\mu}$ and $\boldsymbol{\xi}_{n,i}$, we can remove many of the terms in this equation. Take $\boldsymbol{x} = \boldsymbol{\mu}_+$ as an example, we have

$$
\begin{aligned}
&\boldsymbol{\mu}_+^\top \nabla_{\boldsymbol{W}_V} L_S(\theta) \boldsymbol{w}_O \\
&= \frac{\|\boldsymbol{\mu}\|_2^2}{NM} \sum_{n \in S_+} \ell_n'(\theta) \big( \frac{\exp(\boldsymbol{\mu}_+^\top \boldsymbol{W}_Q \boldsymbol{W}_K^\top \boldsymbol{\mu}_+)}{\exp(\boldsymbol{\mu}_+^\top \boldsymbol{W}_Q \boldsymbol{W}_K^\top \boldsymbol{\mu}_+) + \sum_{k=2}^{M} \exp(\boldsymbol{\mu}_+^\top \boldsymbol{W}_Q \boldsymbol{W}_K^\top \boldsymbol{\xi}_{n,k})} \\
&+ \sum_{j=2}^{M} \frac{\exp(\boldsymbol{\xi}_{n,j}^\top \boldsymbol{W}_Q \boldsymbol{W}_K^\top \boldsymbol{\mu}_+)}{\exp(\boldsymbol{\xi}_{n,j}^\top \boldsymbol{W}_Q \boldsymbol{W}_K^\top \boldsymbol{\mu}_+) + \sum_{k=2}^{M} \exp(\boldsymbol{\xi}_{n,j}^\top \boldsymbol{W}_Q \boldsymbol{W}_K^\top \boldsymbol{\xi}_{n,k})} \big) \|\boldsymbol{w}_O\|_2^2
\end{aligned}
\tag{11}
$$

Then we have

$$
\begin{aligned}
&\boldsymbol{\mu}_+^\top \boldsymbol{W}_V^{(t+1)} \boldsymbol{w}_O - \boldsymbol{\mu}_+^\top \boldsymbol{W}_V^{(t)} \boldsymbol{w}_O = \boldsymbol{\mu}_+^\top \big( -\eta \nabla_{\boldsymbol{W}_V} L_S(\theta(t)) \big) \boldsymbol{w}_O \\
&- \frac{\eta \|\boldsymbol{\mu}\|_2^2}{NM} \sum_{n \in S_+} \ell_n'^{(t)} \big( \frac{\exp(\langle \boldsymbol{q}_+^{(t)}, \boldsymbol{k}_+^{(t)} \rangle)}{\exp(\langle \boldsymbol{q}_+^{(t)}, \boldsymbol{k}_+^{(t)} \rangle) + \sum_{k=2}^{M} \exp(\langle \boldsymbol{q}_+^{(t)}, \boldsymbol{k}_{n,k}^{(t)} \rangle)} \\
&+ \sum_{j=2}^{M} \frac{\exp(\langle \boldsymbol{q}_{n,j}^{(t)}, \boldsymbol{k}_+^{(t)} \rangle)}{\exp(\langle \boldsymbol{q}_{n,j}^{(t)}, \boldsymbol{k}_+^{(t)} \rangle) + \sum_{k=2}^{M} \exp(\langle \boldsymbol{q}_{n,j}^{(t)}, \boldsymbol{k}_{n,k}^{(t)} \rangle)} \big) \|\boldsymbol{w}_O\|_2^2
\end{aligned}
\tag{12}
$$

Dividing by $\|\boldsymbol{w}_O\|_2^2$ we get

$$
\begin{aligned}
&\gamma_{V,+}^{(t+1)} = \gamma_{V,+}^{(t)} \\
&- \frac{\eta \|\boldsymbol{\mu}\|_2^2}{NM} \sum_{n \in S_+} \ell_n'^{(t)} \big( \frac{\exp(\langle \boldsymbol{q}_+^{(t)}, \boldsymbol{k}_+^{(t)} \rangle)}{\exp(\langle \boldsymbol{q}_+^{(t)}, \boldsymbol{k}_+^{(t)} \rangle) + \sum_{k=2}^{M} \exp(\langle \boldsymbol{q}_+^{(t)}, \boldsymbol{k}_{n,k}^{(t)} \rangle)} \\
&+ \sum_{j=2}^{M} \frac{\exp(\langle \boldsymbol{q}_{n,j}^{(t)}, \boldsymbol{k}_+^{(t)} \rangle)}{\exp(\langle \boldsymbol{q}_{n,j}^{(t)}, \boldsymbol{k}_+^{(t)} \rangle) + \sum_{k=2}^{M} \exp(\langle \boldsymbol{q}_{n,j}^{(t)}, \boldsymbol{k}_{n,k}^{(t)} \rangle)} \big)
\end{aligned}
\tag{13}
$$

This proves the update rule for $\gamma_{V,+}^{(t)}$, The proof for $\gamma_{V,+}^{(t)}$ and $\rho_{V,n,i}^{(t)}$ is similar to it.

**Lemma B.3** (Update Rule for QK). *The dynamics of $\boldsymbol{x}^\top \boldsymbol{W}_Q \boldsymbol{W}_K \boldsymbol{x}$ can be characterized as follows*

$$
\begin{aligned}
&\langle \boldsymbol{q}_+^{(t+1)}, \boldsymbol{k}_+^{(t+1)} \rangle - \langle \boldsymbol{q}_+^{(t)}, \boldsymbol{k}_+^{(t)} \rangle \\
&= \alpha_{+,+}^{(t)} \|\boldsymbol{k}_+^{(t)}\|_2^2 + \sum_{n \in S_+} \sum_{i=2}^{M} \alpha_{n,+,i}^{(t)} \langle \boldsymbol{k}_+^{(t)}, \boldsymbol{k}_{n,i}^{(t)} \rangle \\
&+ \beta_{+,+}^{(t)} \|\boldsymbol{q}_+^{(t)}\|_2^2 + \sum_{n \in S_+} \sum_{i=2}^{M} \beta_{n,+,i}^{(t)} \langle \boldsymbol{q}_+^{(t)}, \boldsymbol{q}_{n,i}^{(t)} \rangle \\
&+ \big( \alpha_{+,+}^{(t)} \boldsymbol{k}_+^{(t)} + \sum_{n \in S_+} \sum_{i=2}^{M} \alpha_{n,+,i}^{(t)} \boldsymbol{k}_{n,i}^{(t)} \big) \\
&\cdot \big( \beta_{+,+}^{(t)} \boldsymbol{q}_+^{(t)\top} + \sum_{n \in S_+} \sum_{i=2}^{M} \beta_{n,+,i}^{(t)} \boldsymbol{q}_{n,i}^{(t)\top} \big),
\end{aligned}
$$

$$\langle \boldsymbol{q}_-^{(t+1)}, \boldsymbol{k}_-^{(t+1)}\rangle - \langle \boldsymbol{q}_-^{(t)}, \boldsymbol{k}_-^{(t)}\rangle$$

$$= \alpha_{-,-}^{(t)}\|\boldsymbol{k}_-^{(t)}\|_2^2 + \sum_{n\in S_-}\sum_{i=2}^{M}\alpha_{n,-,i}^{(t)}\langle \boldsymbol{k}_-^{(t)}, \boldsymbol{k}_{n,i}^{(t)}\rangle$$

$$+ \beta_{-,-}^{(t)}\|\boldsymbol{q}_-^{(t)}\|_2^2 + \sum_{n\in S_-}\sum_{i=2}^{M}\beta_{n,-,i}^{(t)}\langle \boldsymbol{q}_-^{(t)}, \boldsymbol{q}_{n,i}^{(t)}\rangle$$

$$+ \left(\alpha_{-,-}^{(t)}\boldsymbol{k}_-^{(t)} + \sum_{n\in S_-}\sum_{i=2}^{M}\alpha_{n,-,i}^{(t)}\boldsymbol{k}_{n,i}^{(t)}\right)$$

$$\cdot \left(\beta_{-,-}^{(t)}\boldsymbol{q}_-^{(t)\top} + \sum_{n\in S_-}\sum_{i=2}^{M}\beta_{n,-,i}^{(t)}\boldsymbol{q}_{n,i}^{(t)\top}\right),$$

$$\langle \boldsymbol{q}_{n,i}^{(t+1)}, \boldsymbol{k}_+^{(t+1)}\rangle - \langle \boldsymbol{q}_{n,i}^{(t)}, \boldsymbol{k}_+^{(t)}\rangle$$

$$= \alpha_{n,i,+}^{(t)}\|\boldsymbol{k}_+^{(t)}\|_2^2 + \alpha_{n,i,-}^{(t)}\langle \boldsymbol{k}_+^{(t)}, \boldsymbol{k}_-^{(t)}\rangle + \sum_{n'=1}^{N}\sum_{l=2}^{M}\alpha_{n,i,n',l}^{(t)}\langle \boldsymbol{k}_+^{(t)}, \boldsymbol{k}_{n',l}^{(t)}\rangle$$

$$+ \beta_{+,+}^{(t)}\langle \boldsymbol{q}_+^{(t)}, \boldsymbol{q}_{n,i}^{(t)}\rangle + \sum_{n'\in S_+}\sum_{l=2}^{M}\beta_{n',+,l}^{(t)}\langle \boldsymbol{q}_{n,i}^{(t)}, \boldsymbol{q}_{n',l}^{(t)}\rangle$$

$$+ \left(\alpha_{n,i,+}^{(t)}\boldsymbol{k}_+^{(t)} + \alpha_{n,i,-}^{(t)}\boldsymbol{k}_-^{(t)} + \sum_{n'=1}^{N}\sum_{l=2}^{M}\alpha_{n,i,n',l}^{(t)}\boldsymbol{k}_{n',l}^{(t)}\right)$$

$$\cdot \left(\beta_{+,+}^{(t)}\boldsymbol{q}_+^{(t)\top} + \sum_{n'\in S_+}\sum_{l=2}^{M}\beta_{n',+,l}^{(t)}\boldsymbol{q}_{n',l}^{(t)\top}\right),$$

$$\langle \boldsymbol{q}_{n,i}^{(t+1)}, \boldsymbol{k}_-^{(t+1)}\rangle - \langle \boldsymbol{q}_{n,i}^{(t)}, \boldsymbol{k}_-^{(t)}\rangle$$

$$= \alpha_{n,i,-}^{(t)}\|\boldsymbol{k}_-^{(t)}\|_2^2 + \alpha_{n,i,+}^{(t)}\langle \boldsymbol{k}_+^{(t)}, \boldsymbol{k}_-^{(t)}\rangle + \sum_{n'=1}^{N}\sum_{l=2}^{M}\alpha_{n,i,n',l}^{(t)}\langle \boldsymbol{k}_-^{(t)}, \boldsymbol{k}_{n',l}^{(t)}\rangle$$

$$+ \beta_{-,-}^{(t)}\langle \boldsymbol{q}_-^{(t)}, \boldsymbol{q}_{n,i}^{(t)}\rangle + \sum_{n'\in S_-}\sum_{l=2}^{M}\beta_{n',-,l}^{(t)}\langle \boldsymbol{q}_{n,i}^{(t)}, \boldsymbol{q}_{n',l}^{(t)}\rangle$$

$$+ \left(\alpha_{n,i,+}^{(t)}\boldsymbol{k}_+^{(t)} + \alpha_{n,i,-}^{(t)}\boldsymbol{k}_-^{(t)} + \sum_{n'=1}^{N}\sum_{l=2}^{M}\alpha_{n,i,n',l}^{(t)}\boldsymbol{k}_{n',l}^{(t)}\right)$$

$$\cdot \left(\beta_{-,-}^{(t)}\boldsymbol{q}_-^{(t)\top} + \sum_{n'\in S_-}\sum_{l=2}^{M}\beta_{n',-,l}^{(t)}\boldsymbol{q}_{n',l}^{(t)\top}\right),$$

$$\langle \boldsymbol{q}_+^{(t+1)}, \boldsymbol{k}_{n,j}^{(t+1)} \rangle - \langle \boldsymbol{q}_+^{(t)}, \boldsymbol{k}_{n,j}^{(t)} \rangle$$

$$= \alpha_{+,+}^{(t)} \langle \boldsymbol{k}_+^{(t)}, \boldsymbol{k}_{n,j}^{(t)} \rangle + \sum_{n' \in S_+} \sum_{l=2}^{M} \alpha_{n',+,l}^{(t)} \langle \boldsymbol{k}_{n,j}^{(t)}, \boldsymbol{k}_{n',l}^{(t)} \rangle$$

$$+ \beta_{n,j,+}^{(t)} \|\boldsymbol{q}_+^{(t)}\|_2^2 + \beta_{n,j,-}^{(t)} \langle \boldsymbol{q}_+^{(t)}, \boldsymbol{q}_-^{(t)} \rangle + \sum_{n'=1}^{N} \sum_{l=2}^{M} \beta_{n,j,n',l}^{(t)} \langle \boldsymbol{q}_+^{(t)}, \boldsymbol{q}_{n',l}^{(t)} \rangle$$

$$+ \left( \alpha_{+,+}^{(t)} \boldsymbol{k}_+^{(t)} + \sum_{n' \in S_+} \sum_{l=2}^{M} \alpha_{n',+,l}^{(t)} \boldsymbol{k}_{n',l}^{(t)} \right)$$

$$\cdot \left( \beta_{n,j,+}^{(t)} \boldsymbol{q}_+^{(t)\top} + \beta_{n,j,-}^{(t)} \boldsymbol{q}_-^{(t)\top} + \sum_{n'=1}^{N} \sum_{l=2}^{M} \beta_{n,j,n',l}^{(t)} \boldsymbol{q}_{n',l}^{(t)\top} \right),$$

$$\langle \boldsymbol{q}_-^{(t+1)}, \boldsymbol{k}_{n,j}^{(t+1)} \rangle - \langle \boldsymbol{q}_-^{(t)}, \boldsymbol{k}_{n,j}^{(t)} \rangle$$

$$= \alpha_{-,-}^{(t)} \langle \boldsymbol{k}_-^{(t)}, \boldsymbol{k}_{n,j}^{(t)} \rangle + \sum_{n' \in S_-} \sum_{l=2}^{M} \alpha_{n',-,l}^{(t)} \langle \boldsymbol{k}_{n,j}^{(t)}, \boldsymbol{k}_{n',l}^{(t)} \rangle$$

$$+ \beta_{n,j,-}^{(t)} \|\boldsymbol{q}_-^{(t)}\|_2^2 + \beta_{n,j,+}^{(t)} \langle \boldsymbol{q}_+^{(t)}, \boldsymbol{q}_-^{(t)} \rangle + \sum_{n'=1}^{N} \sum_{l=2}^{M} \beta_{n,j,n',l}^{(t)} \langle \boldsymbol{q}_-^{(t)}, \boldsymbol{q}_{n',l}^{(t)} \rangle$$

$$+ \left( \alpha_{-,-}^{(t)} \boldsymbol{k}_-^{(t)} + \sum_{n' \in S_-} \sum_{l=2}^{M} \alpha_{n',-,l}^{(t)} \boldsymbol{k}_{n',l}^{(t)} \right)$$

$$\cdot \left( \beta_{n,j,+}^{(t)} \boldsymbol{q}_+^{(t)\top} + \beta_{n,j,-}^{(t)} \boldsymbol{q}_-^{(t)\top} + \sum_{n'=1}^{N} \sum_{l=2}^{M} \beta_{n,j,n',l}^{(t)} \boldsymbol{q}_{n',l}^{(t)\top} \right),$$

$$\langle \boldsymbol{q}_{n,i}^{(t+1)}, \boldsymbol{k}_{n,j}^{(t+1)} \rangle - \langle \boldsymbol{q}_{n,i}^{(t)}, \boldsymbol{k}_{n,j}^{(t)} \rangle$$

$$= \alpha_{n,i,+}^{(t)} \langle \boldsymbol{k}_+^{(t)}, \boldsymbol{k}_{n,j}^{(t)} \rangle + \alpha_{n,i,-}^{(t)} \langle \boldsymbol{k}_-^{(t)}, \boldsymbol{k}_{n,j}^{(t)} \rangle + \sum_{n'=1}^{N} \sum_{l=2}^{M} \alpha_{n,i,n',l}^{(t)} \langle \boldsymbol{k}_{n',l}^{(t)}, \boldsymbol{k}_{n,j}^{(t)} \rangle$$

$$+ \beta_{n,j,+}^{(t)} \langle \boldsymbol{q}_+^{(t)}, \boldsymbol{q}_{n,i}^{(t)} \rangle + \beta_{n,j,-}^{(t)} \langle \boldsymbol{q}_-^{(t)}, \boldsymbol{q}_{n,i}^{(t)} \rangle + \sum_{n'=1}^{N} \sum_{l=2}^{M} \beta_{n,j,n',l}^{(t)} \langle \boldsymbol{q}_{n',l}^{(t)}, \boldsymbol{q}_{n,i}^{(t)} \rangle$$

$$+ \left( \alpha_{n,i,+}^{(t)} \boldsymbol{k}_+^{(t)} + \alpha_{n,i,-}^{(t)} \boldsymbol{k}_-^{(t)} + \sum_{n'=1}^{N} \sum_{l=2}^{M} \alpha_{n,i,n',l}^{(t)} \boldsymbol{k}_{n',l}^{(t)} \right)$$

$$\cdot \left( \beta_{n,j,+}^{(t)} \boldsymbol{q}_+^{(t)\top} + \beta_{n,j,-}^{(t)} \boldsymbol{q}_-^{(t)\top} + \sum_{n'=1}^{N} \sum_{l=2}^{M} \beta_{n,j,n',l}^{(t)} \boldsymbol{q}_{n',l}^{(t)\top} \right),$$

for $i, j \in [M] \backslash \{1\}, n \in [N]$.

*Proof of Lemma B.3.* Based on Definition B.3, we have

$$\Delta \boldsymbol{q}_+^{(t)} := \boldsymbol{q}_+^{(t+1)} - \boldsymbol{q}_+^{(t)} = \alpha_{+,+}^{(t)} \boldsymbol{k}_+^{(t)} + \sum_{n \in S_+} \sum_{i=2}^{M} \alpha_{n,+,i}^{(t)} \boldsymbol{k}_{n,i}^{(t)} \tag{14}$$

$$\Delta \boldsymbol{k}_+^{(t)} := \boldsymbol{k}_+^{(t+1)} - \boldsymbol{k}_+^{(t)} = \beta_{+,+}^{(t)} \boldsymbol{q}_+^{(t)} + \sum_{n \in S_+} \sum_{i=2}^{M} \beta_{n,+,i}^{(t)} \boldsymbol{q}_{n,i}^{(t)} \tag{15}$$

Then we get

$$
\begin{aligned}
&\langle q_+^{(t+1)}, k_+^{(t+1)} \rangle - \langle q_+^{(t)}, k_+^{(t)} \rangle \\
&= \mu_+^\top (W_Q^{(t)} + \Delta W_Q^{(t)})(W_K^{(t)\top} + \Delta W_K^{(t)\top})\mu_+ - \langle q_+^{(t)}, k_+^{(t)} \rangle \\
&= \langle \Delta q_+^{(t)}, k_+^{(t)} \rangle + \langle q_+^{(t)}, \Delta k_+^{(t)} \rangle + \langle \Delta q_+^{(t)}, \Delta k_+^{(t)} \rangle \\
&= \alpha_{+,+}^{(t)} \| k_+^{(t)} \|_2^2 + \sum_{n \in S_+} \sum_{i=2}^{M} \alpha_{n,+,i}^{(t)} \langle k_+^{(t)}, k_{n,i}^{(t)} \rangle \\
&\quad + \beta_{+,+}^{(t)} \| q_+^{(t)} \|_2^2 + \sum_{n \in S_+} \sum_{i=2}^{M} \beta_{n,+,i}^{(t)} \langle q_+^{(t)}, q_{n,i}^{(t)} \rangle \\
&\quad + \left( \alpha_{+,+}^{(t)} k_+^{(t)} + \sum_{n \in S_+} \sum_{i=2}^{M} \alpha_{n,+,i}^{(t)} k_{n,i}^{(t)} \right) \\
&\quad \cdot \left( \beta_{+,+}^{(t)} q_+^{(t)\top} + \sum_{n \in S_+} \sum_{i=2}^{M} \beta_{n,+,i}^{(t)} q_{n,i}^{(t)\top} \right)
\end{aligned}
\tag{16}
$$

where $\Delta W_Q^{(t)} := W_Q^{(t+1)} - W_Q^{(t)}$, $\Delta W_K^{(t)} := W_K^{(t+1)} - W_K^{(t)}$, the second equality is by $\Delta q_+^{(t)} = \mu_+^\top \Delta W_Q^{(t)}$ and $\Delta k_+^{(t)} = \mu_+^\top \Delta W_K^{(t)}$, the third equality is by plugging (14) and (15). This proves the update rule for $\langle q_+^{(t)}, k_+^{(t)} \rangle$, the proof for $\langle q_{n,i}^{(t)}, k_+^{(t)} \rangle, \langle q_+^{(t)}, k_{n,j}^{(t)} \rangle, \langle q_{n,i}^{(t)}, k_{n,j}^{(t)} \rangle$ are exactly the same.

**Remark.** In (16), note that the magnitude of $\langle k_+^{(t)}, k_{n,i}^{(t)} \rangle, \langle q_+^{(t)}, q_{n,i}^{(t)} \rangle$ is much smaller than that of $\| k_+^{(t)} \|_2^2, \| q_+^{(t)} \|_2^2$, and that the magnitude of $\alpha_{n,+,i}^{(t)}, \beta_{n,+,i}^{(t)}$ will not be greater than that of $\alpha_{+,+}^{(t)}, \beta_{+,+}^{(t)}$ (we will prove it later). Also, since the learning rate $\eta$ is sufficiently small, the last term of (16) will be very small. Then (16) can be expressed in the following form

$$
\begin{aligned}
&\langle q_+^{(t+1)}, k_+^{(t+1)} \rangle - \langle q_+^{(t)}, k_+^{(t)} \rangle \\
&= \alpha_{+,+}^{(t)} \| k_+^{(t)} \|_2^2 + \beta_{+,+}^{(t)} \| q_+^{(t)} \|_2^2 + \{lower\ order\ term\}
\end{aligned}
$$

The dynamics of $\langle q_{n,i}^{(t)}, k_+^{(t)} \rangle, \langle q_+^{(t)}, k_{n,j}^{(t)} \rangle, \langle q_{n,i}^{(t)}, k_{n,j}^{(t)} \rangle$ can be expressed in a similar way. Later we will rigorously explain the so-call {lower order term}.

## C  Concentration Inequalities

In this section, we will give some concentration inequalities that show some important properties of the data and the ViT parameters at random initialization.

**Lemma C.1** (Lemma B.1 in Cao et al. (2022)). *Suppose that $\delta > 0$ and $n \geq 8 \log(4/\delta)$. Then with probability at least $1 - \delta$,*

$$
\frac{N}{4} \leq |\{n \in [N] : y_n = 1\}|, |\{n \in [N] : y_n = -1\}| \leq \frac{3N}{4}.
$$

**Lemma C.2** (Initialization of V). *Suppose that $\delta > 0$. Then with probability at least $1 - \delta$,*

$$
|V_\pm^{(0)}| \leq d_h^{-\frac{1}{4}},
$$

$$
|V_{n,i}^{(0)}| \leq d_h^{-\frac{1}{4}}
$$

*for $i \in [M] \backslash \{1\}, n \in [N]$.*

*Proof of Lemma C.2.* It is clear that $\mu_\pm^\top W_V^{(0)} w_O$ is a random variable with mean zero and variance $\sigma_V^2 \| w_O \|_2^2 \| \mu \|_2^2$. Therefore, by Gaussian tail bound and the condition that $\sigma_V \leq \widetilde{O}(\| w_O \|_2^{-1} \cdot \min\{\| \mu \|_2^{-1}, (\sigma_p \sqrt{d})^{-1}\} \cdot d_h^{-\frac{1}{4}})$, with probability at least $1 - \delta/NM$,

$$
|\mu_\pm^\top W_V^{(0)} w_O| \leq \sigma_V \| w_O \|_2 \| \mu \|_2 \sqrt{\log(2NM/\delta)} \leq d_h^{-\frac{1}{4}}
$$

Moreover, each element of vector $\boldsymbol{w}_O^\top \boldsymbol{W}_V^{(0)}$ is a random variable with mean zero and variance $\sigma_V^2 \|\boldsymbol{w}_O\|_2^2$. Therefore, by Bernstein's inequality and the condition of $\sigma_V$, with probability at least $1 - \delta/NM$

$$|\boldsymbol{\xi}_{n,i}^\top \boldsymbol{W}_V^{(0)} \boldsymbol{w}_O| \leq 2\sigma_V \sigma_p \|\boldsymbol{w}_O\|_2 \sqrt{d \log(2NM/\delta)} \leq d_h^{-\frac{1}{4}}$$

Applying a union bound completes the proof.

**Lemma C.3** (Initialization of QK). *Suppose that $\delta > 0$. Then with probability at least $1 - \delta$,*

$$\frac{\|\boldsymbol{\mu}\|_2^2 \sigma_h^2 d_h}{2} \leq \|\boldsymbol{q}_\pm^{(0)}\|_2^2 \leq \frac{3\|\boldsymbol{\mu}\|_2^2 \sigma_h^2 d_h}{2},$$

$$\frac{\tilde{\sigma}_p^2 \sigma_h^2 d d_h}{2} \leq \|\boldsymbol{q}_{n,2}^{(0)}\|_2^2 \leq \frac{3\tilde{\sigma}_p^2 \sigma_h^2 d d_h}{2},$$

$$\frac{\sigma_p^2 \sigma_h^2 d d_h}{2} \leq \|\boldsymbol{q}_{n,i'}^{(0)}\|_2^2 \leq \frac{3\sigma_p^2 \sigma_h^2 d d_h}{2},$$

$$\frac{\|\boldsymbol{\mu}\|_2^2 \sigma_h^2 d_h}{2} \leq \|\boldsymbol{k}_\pm^{(0)}\|_2^2 \leq \frac{3\|\boldsymbol{\mu}\|_2^2 \sigma_h^2 d_h}{2},$$

$$\frac{\tilde{\sigma}_p^2 \sigma_h^2 d d_h}{2} \leq \|\boldsymbol{k}_{n,2}^{(0)}\|_2^2 \leq \frac{3\tilde{\sigma}_p^2 \sigma_h^2 d d_h}{2},$$

$$\frac{\sigma_p^2 \sigma_h^2 d d_h}{2} \leq \|\boldsymbol{k}_{n,i'}^{(0)}\|_2^2 \leq \frac{3\sigma_p^2 \sigma_h^2 d d_h}{2},$$

$$|\langle \boldsymbol{q}_+^{(0)}, \boldsymbol{q}_-^{(0)} \rangle| \leq 2\|\boldsymbol{\mu}\|_2^2 \sigma_h^2 \cdot \sqrt{d_h \log(6N^2 M^2/\delta)},$$

$$|\langle \boldsymbol{q}_\pm^{(0)}, \boldsymbol{q}_{n,i}^{(0)} \rangle| \leq 2\|\boldsymbol{\mu}\|_2 \tilde{\sigma}_p \sigma_h^2 d^{\frac{1}{2}} \cdot \sqrt{d_h \log(6N^2 M^2/\delta)},$$

$$|\langle \boldsymbol{k}_+^{(0)}, \boldsymbol{k}_-^{(0)} \rangle| \leq 2\|\boldsymbol{\mu}\|_2^2 \sigma_h^2 \cdot \sqrt{d_h \log(6N^2 M^2/\delta)},$$

$$|\langle \boldsymbol{q}_\pm^{(0)}, \boldsymbol{k}_\pm^{(0)} \rangle| \leq 2\|\boldsymbol{\mu}\|_2^2 \sigma_h^2 \cdot \sqrt{d_h \log(6N^2 M^2/\delta)},$$

$$|\langle \boldsymbol{q}_\pm^{(0)}, \boldsymbol{k}_\mp^{(0)} \rangle| \leq 2\|\boldsymbol{\mu}\|_2^2 \sigma_h^2 \cdot \sqrt{d_h \log(6N^2 M^2/\delta)},$$

$$|\langle \boldsymbol{q}_{n,i}^{(0)}, \boldsymbol{k}_\pm^{(0)} \rangle| \leq 2\|\boldsymbol{\mu}\|_2 \tilde{\sigma}_p \sigma_h^2 d^{\frac{1}{2}} \cdot \sqrt{d_h \log(6N^2 M^2/\delta)},$$

$$|\langle \boldsymbol{q}_{n,i}^{(0)}, \boldsymbol{q}_{n',j}^{(0)} \rangle| \leq 2\tilde{\sigma}_p^2 \sigma_h^2 d \cdot \sqrt{d_h \log(6N^2 M^2/\delta)},$$

$$|\langle \boldsymbol{k}_{n,i}^{(0)}, \boldsymbol{k}_{n',j}^{(0)} \rangle| \leq 2\tilde{\sigma}_p^2 \sigma_h^2 d \cdot \sqrt{d_h \log(6N^2 M^2/\delta)},$$

$$|\langle \boldsymbol{k}_\pm^{(0)}, \boldsymbol{k}_{n,i}^{(0)} \rangle| \leq 2\|\boldsymbol{\mu}\|_2 \tilde{\sigma}_p \sigma_h^2 d^{\frac{1}{2}} \cdot \sqrt{d_h \log(6N^2 M^2/\delta)},$$

$$|\langle \boldsymbol{q}_\pm^{(0)}, \boldsymbol{k}_{n,i}^{(0)} \rangle| \leq 2\|\boldsymbol{\mu}\|_2 \tilde{\sigma}_p \sigma_h^2 d^{\frac{1}{2}} \cdot \sqrt{d_h \log(6N^2 M^2/\delta)},$$

$$|\langle \boldsymbol{q}_{n,i}^{(0)}, \boldsymbol{k}_{n',j}^{(0)} \rangle| \leq 2\tilde{\sigma}_p^2 \sigma_h^2 d \cdot \sqrt{d_h \log(6N^2 M^2/\delta)}$$

*for $i, j \in [M] \backslash \{1\}$, $i' \in [M] \backslash \{1, 2\}$ and $n, n' \in [N]$.*

*Proof of Lemma C.3.* It is clear that each element of vector $\boldsymbol{q}_+^{(0)}$ is a random variable with mean zero and variance $\|\boldsymbol{\mu}\|_2^2 \sigma_h^2$. By Bernstein's inequality, with probability at least $1 - \delta/3N^2 M^2$, we have

$$\big| \|\boldsymbol{q}_+^{(0)}\|_2^2 - \|\boldsymbol{\mu}\|_2^2 \sigma_h^2 d_h \big| = O(\|\boldsymbol{\mu}\|_2^2 \sigma_h^2 \sqrt{d_h \log(6N^2 M^2/\delta)}).$$

Therefore, as long as $d_h = \Omega(\log(6N^2 M^2/\delta))$, we have

$$\frac{\|\boldsymbol{\mu}\|_2^2 \sigma_h^2 d_h}{2} \leq \|\boldsymbol{q}_+^{(0)}\|_2^2 \leq \frac{3\|\boldsymbol{\mu}\|_2^2 \sigma_h^2 d_h}{2},$$

Similarly, we have

$$\frac{\|\boldsymbol{\mu}\|_2^2 \sigma_h^2 d_h}{2} \leq \|\boldsymbol{q}_-^{(0)}\|_2^2 \leq \frac{3\|\boldsymbol{\mu}\|_2^2 \sigma_h^2 d_h}{2},$$

$$\frac{\tilde{\sigma}_p^2 \sigma_h^2 d d_h}{2} \leq \|\boldsymbol{q}_{n,2}^{(0)}\|_2^2 \leq \frac{3\tilde{\sigma}_p^2 \sigma_h^2 d d_h}{2},$$

$$\frac{\sigma_p^2 \sigma_h^2 dd_h}{2} \le \|\boldsymbol{q}_{n,i'}^{(0)}\|_2^2 \le \frac{3\sigma_p^2 \sigma_h^2 dd_h}{2},$$

$$\frac{\|\boldsymbol{\mu}\|_2^2 \sigma_h^2 d_h}{2} \le \|\boldsymbol{k}_\pm^{(0)}\|_2^2 \le \frac{3\|\boldsymbol{\mu}\|_2^2 \sigma_h^2 d_h}{2},$$

$$\frac{\tilde{\sigma}_p^2 \sigma_h^2 dd_h}{2} \le \|\boldsymbol{k}_{n,2}^{(0)}\|_2^2 \le \frac{3\tilde{\sigma}_p^2 \sigma_h^2 dd_h}{2},$$

$$\frac{\sigma_p^2 \sigma_h^2 dd_h}{2} \le \|\boldsymbol{k}_{n,i'}^{(0)}\|_2^2 \le \frac{3\sigma_p^2 \sigma_h^2 dd_h}{2},$$

Moreover, $\langle \boldsymbol{q}_+^{(0)}, \boldsymbol{q}_-^{(0)} \rangle$ has mean zero. By Bernstein's inequality, with probability at least $1 - \delta/3N^2M^2$, we have

$$|\langle \boldsymbol{q}_+^{(0)}, \boldsymbol{q}_-^{(0)} \rangle| \le 2\|\boldsymbol{\mu}\|_2^2 \sigma_h^2 \cdot \sqrt{d_h \log(6N^2M^2/\delta)}.$$

We can prove the rest of this lemma in a similar way.

**Lemma C.4** (Lemma B.2 in Cao et al. (2022)). *Suppose that $\delta > 0$ and $d = \Omega(\log(4NM/\delta))$. Then with probability at least $1 - \delta$*

$$\tilde{\sigma}_p^2 d/2 \le \|\boldsymbol{\xi}_{n,2}\|_2^2 \le 3\tilde{\sigma}_p^2 d/2,$$

$$\sigma_p^2 d/2 \le \|\boldsymbol{\xi}_{n,i}\|_2^2 \le 3\sigma_p^2 d/2,$$

$$|\langle \boldsymbol{\xi}_{n,i}, \boldsymbol{\xi}_{n',i'} \rangle| \le 2\tilde{\sigma}_p^2 \cdot \sqrt{d \log(4N^2M^2/\delta)}$$

*for $i, i' \in [M]\backslash\{1\}, n, n' \in [N], i \ne i'$ or $n \ne n'$.*

## D    Benign Overfitting

In this section, we consider the benign overfitting regime under the condition that $N \cdot \mathrm{SNR}^2 = \Omega(1)$. We analyze the dynamics of $V_\pm$, $V_{n,i}$, the inner product of $\boldsymbol{q}_\pm$, $\boldsymbol{q}_{n,i}$ and $\boldsymbol{k}_\pm$, $\boldsymbol{k}_{n,i}$ during gradient descent training, and further give the upper bound for population loss. The proofs in this section are based on the results in Section C, which hold with high probability.

### D.1    Stage I

In Stage I, $V_\pm^{(t)}$, $V_{n,i}^{(t)}$ begin to pull apart until $|V_\pm^{(t)}|$ is sufficiently larger than $|V_{n,i}^{(t)}|$. At the same time, the inner products of $\boldsymbol{q}$ and $\boldsymbol{k}$ maintain their magnitude.

**Lemma D.1** (Gradient of Loss). *As long as $\max\{|V_+^{(t)}|, |V_-^{(t)}|, |V_{n,i}^{(t)}|\} = o(1)$, we have $-\ell'(y_n f(\boldsymbol{X}_n, \theta(t)))$ remains $1/2 \pm o(1)$.*

*Proof of Lemma D.1.* Note that $\ell(z) = \log(1 + \exp(-z))$ and $-\ell' = \exp(-z)/(1 + \exp(-z))$, without loss of generality, we assume $y_n = 1$, we have

$$-\ell'(f(\boldsymbol{X}_n, \theta(t))) = \frac{1}{1 + \exp(\frac{1}{M}\sum_{l=1}^{M} \boldsymbol{\varphi}(\boldsymbol{x}_{n,l}^\top \boldsymbol{W}_Q^{(t)} \boldsymbol{W}_K^{(t)\top} \boldsymbol{X}_n^\top)\boldsymbol{X}_n \boldsymbol{W}_V^{(t)} \boldsymbol{w}_O)}.$$

Note that

$$-\max\{|V_+^{(t)}|, |V_-^{(t)}|, |V_{n,i}^{(t)}|\} \le \frac{1}{M}\sum_{l=1}^{M} \boldsymbol{\varphi}(\boldsymbol{x}_{n,l}^\top \boldsymbol{W}_Q^{(t)} \boldsymbol{W}_K^{(t)\top} \boldsymbol{X}_n^\top)\boldsymbol{X}_n \boldsymbol{W}_V^{(t)} \boldsymbol{w}_O \le \max\{|V_+^{(t)}|, |V_-^{(t)}|, |V_{n,i}^{(t)}|\}.$$

Then we have

$$-\ell'(f(\boldsymbol{X}_n, \theta(t))) \ge \frac{1}{1 + \exp(0 + o(1))} \ge \frac{1}{2 + o(1)} \ge \frac{1}{2} - o(1),$$

$$-\ell'(f(\boldsymbol{X}_n, \theta(t))) \le \frac{\exp(0 + o(1))}{1 + \exp(0 + o(1))} \le \frac{1 + o(1)}{1 + 1 + o(1)} \le \frac{1}{2} + o(1).$$

**Lemma D.2** (Bound of Attention). *As long as* $|\langle \boldsymbol{q}_{\pm}^{(t)}, \boldsymbol{k}_{\pm}^{(t)} \rangle|, |\langle \boldsymbol{q}_{n,i}^{(t)}, \boldsymbol{k}_{\pm}^{(t)} \rangle|, |\langle \boldsymbol{q}_{\pm}^{(t)}, \boldsymbol{k}_{n,j}^{(t)} \rangle|, |\langle \boldsymbol{q}_{n,i}^{(t)}, \boldsymbol{k}_{n,j}^{(t)} \rangle| = o(1)$, *we have*

$$\frac{1}{M} - o(1) \leq softmax(\langle \boldsymbol{q}_{\pm}^{(t)}, \boldsymbol{k}_{\pm}^{(t)} \rangle) \leq \frac{1}{M} + o(1),$$

$$\frac{1}{M} - o(1) \leq softmax(\langle \boldsymbol{q}_{n,i}^{(t)}, \boldsymbol{k}_{\pm}^{(t)} \rangle) \leq \frac{1}{M} + o(1),$$

$$\frac{1}{M} - o(1) \leq softmax(\langle \boldsymbol{q}_{\pm}^{(t)}, \boldsymbol{k}_{n,j}^{(t)} \rangle) \leq \frac{1}{M} + o(1),$$

$$\frac{1}{M} - o(1) \leq softmax(\langle \boldsymbol{q}_{n,i}^{(t)}, \boldsymbol{k}_{n,j}^{(t)} \rangle) \leq \frac{1}{M} + o(1).$$

*Proof of Lemma D.2.* It is clear that $\exp(o(1)) = 1 + o(1)$. Therefore, as long as $|\langle \boldsymbol{q}_{\pm}^{(t)}, \boldsymbol{k}_{\pm}^{(t)} \rangle| = o(1)$, we have

$$\frac{1}{M} - o(1) = \frac{1}{1 + (M-1) + (M-1)o(1)} = \frac{1}{1 + (M-1)\exp(o(1))} =$$

$$\frac{\exp(-o(1))}{\exp(-o(1)) + (M-1)\exp(o(1))} \leq softmax(\langle \boldsymbol{q}_{\pm}^{(t)}, \boldsymbol{k}_{\pm}^{(t)} \rangle) \leq \frac{\exp(o(1))}{\exp(o(1)) + (M-1)\exp(-o(1))}$$

$$= \frac{\exp(o(1))}{\exp(o(1)) + (M-1)} = \frac{1 + o(1)}{1 + o(1) + (M-1)} = \frac{1}{M} + o(1)$$

Similarly, we have

$$\frac{1}{M} - o(1) \leq softmax(\langle \boldsymbol{q}_{n,i}^{(t)}, \boldsymbol{k}_{\pm}^{(t)} \rangle) \leq \frac{1}{M} + o(1),$$

$$\frac{1}{M} - o(1) \leq softmax(\langle \boldsymbol{q}_{\pm}^{(t)}, \boldsymbol{k}_{n,j}^{(t)} \rangle) \leq \frac{1}{M} + o(1),$$

$$\frac{1}{M} - o(1) \leq softmax(\langle \boldsymbol{q}_{n,i}^{(t)}, \boldsymbol{k}_{n,j}^{(t)} \rangle) \leq \frac{1}{M} + o(1).$$

**Lemma D.3** (Upper bound of V). *Let* $T_0 = O(\frac{1}{\eta d_h^{\frac{1}{4}} \|\boldsymbol{\mu}\|_2^2 \|\boldsymbol{w}_O\|_2^2})$. *Then under the same conditions as Theorem 4.1, we have*

$$|V_+^{(t)}|, |V_-^{(t)}|, |V_{n,i}^{(t)}| = O(d_h^{-\frac{1}{4}})$$

*for* $t \in [0, T_0]$.

*Proof of Lemma D.3.* By Lemma B.2, we have

$$|\gamma_{V,+}^{(t+1)} - \gamma_{V,+}^{(t)}|$$

$$\leq -\frac{\eta \|\boldsymbol{\mu}\|_2^2}{NM} \sum_{n \in S_+} \ell_n'^{(t)} \Big( \frac{\exp(\langle \boldsymbol{q}_+^{(t)}, \boldsymbol{k}_+^{(t)} \rangle)}{\exp(\langle \boldsymbol{q}_+^{(t)}, \boldsymbol{k}_+^{(t)} \rangle) + \sum_{k=2}^{M} \exp(\langle \boldsymbol{q}_+^{(t)}, \boldsymbol{k}_{n,k}^{(t)} \rangle)}$$

$$+ \sum_{j=2}^{M} \frac{\exp(\langle \boldsymbol{q}_{n,j}^{(t)}, \boldsymbol{k}_+^{(t)} \rangle)}{\exp(\langle \boldsymbol{q}_{n,j}^{(t)}, \boldsymbol{k}_+^{(t)} \rangle) + \sum_{k=2}^{M} \exp(\langle \boldsymbol{q}_{n,j}^{(t)}, \boldsymbol{k}_{n,k}^{(t)} \rangle)} \Big)$$

$$\leq \frac{\eta \|\boldsymbol{\mu}\|_2^2}{NM} \cdot \frac{3N}{4} \big( 1 + (M-1) \big)$$

$$\leq \frac{3\eta \|\boldsymbol{\mu}\|_2^2}{4},$$

where the second inequality is by Lemma C.1 and $-\ell_n'^{(t)} \leq 1$. Similarly, we have

$$|\gamma_{V,-}^{(t+1)} - \gamma_{V,-}^{(t)}| \leq \frac{3\eta \|\boldsymbol{\mu}\|_2^2}{4}.$$

By Definition B.1, we have

$$\begin{aligned}
|V_+^{(t)}| &= \Big|V_+^{(0)} + \sum_{s=0}^{t-1}(\gamma_{V,+}^{(s+1)} - \gamma_{V,+}^{(s)})\|\boldsymbol{w}_O\|_2^2\Big| \\
&\leq |V_+^{(0)}| + \sum_{s=0}^{t-1}|\gamma_{V,+}^{(s+1)} - \gamma_{V,+}^{(s)}| \cdot \|\boldsymbol{w}_O\|_2^2 \\
&\leq d_h^{-\frac{1}{4}} + \frac{3\eta\|\boldsymbol{\mu}\|_2^2}{4} \cdot \|\boldsymbol{w}_O\|_2^2 \cdot O(\frac{1}{\eta d_h^{\frac{1}{4}}\|\boldsymbol{\mu}\|_2^2\|\boldsymbol{w}_O\|_2^2}) \\
&= O(d_h^{-\frac{1}{4}}),
\end{aligned}$$

where the first inequality is by triangle inequality, the second inequality is by Lemma C.2. Similarly, we have $|V_-^{(t)}| = O(d_h^{-\frac{1}{4}})$.

By Lemma B.2, we have

$$\begin{aligned}
&|\rho_{V,n,i}^{(t+1)} - \rho_{V,n,i}^{(t)}| \\
&\leq \Big| -\frac{\eta}{NM}\sum_{n'\in S_+}\ell_{n'}^{\prime(t)}\Big(\sum_{i=2}^{M}(\langle\boldsymbol{\xi}_{n,i},\boldsymbol{\xi}_{n',i'}\rangle \frac{\exp(\langle\boldsymbol{q}_+^{(t)},\boldsymbol{k}_{n',i'}^{(t)}\rangle)}{\exp(\langle\boldsymbol{q}_+^{(t)},\boldsymbol{k}_+^{(t)}\rangle) + \sum\limits_{k=2}^{M}\exp(\langle\boldsymbol{q}_+^{(t)},\boldsymbol{k}_{n',k}^{(t)}\rangle)} \\
&\quad + \sum_{j=2}^{M}\langle\boldsymbol{\xi}_{n,i},\boldsymbol{\xi}_{n',i'}\rangle\frac{\exp(\langle\boldsymbol{q}_{n',j}^{(t)},\boldsymbol{k}_{n',i'}^{(t)}\rangle)}{\exp(\langle\boldsymbol{q}_{n',j}^{(t)},\boldsymbol{k}_+^{(t)}\rangle) + \sum\limits_{k=2}^{M}\exp(\langle\boldsymbol{q}_{n',j}^{(t)},\boldsymbol{k}_{n',k}^{(t)}\rangle)}\Big)\Big) \\
&\quad + \frac{\eta}{NM}\sum_{n'\in S_-}\ell_{n'}^{\prime(t)}\Big(\sum_{i=2}^{M}(\langle\boldsymbol{\xi}_{n,i},\boldsymbol{\xi}_{n',i'}\rangle\frac{\exp(\langle\boldsymbol{q}_-^{(t)},\boldsymbol{k}_{n',i'}^{(t)}\rangle)}{\exp(\langle\boldsymbol{q}_-^{(t)},\boldsymbol{k}_-^{(t)}\rangle) + \sum\limits_{k=2}^{M}\exp(\langle\boldsymbol{q}_-^{(t)},\boldsymbol{k}_{n',k}^{(t)}\rangle)} \quad\quad (17) \\
&\quad + \sum_{j=2}^{M}\langle\boldsymbol{\xi}_{n,i},\boldsymbol{\xi}_{n',i'}\rangle\frac{\exp(\langle\boldsymbol{q}_{n',j}^{(t)},\boldsymbol{k}_{n',i'}^{(t)}\rangle)}{\exp(\langle\boldsymbol{q}_{n',j}^{(t)},\boldsymbol{k}_-^{(t)}\rangle) + \sum\limits_{k=2}^{M}\exp(\langle\boldsymbol{q}_{n',j}^{(t)},\boldsymbol{k}_{n',k}^{(t)}\rangle)}\Big)\Big)\Big| \\
&\leq \frac{3\eta\tilde{\sigma}_p^2 d}{2NM}\cdot M + \frac{\eta}{NM}\cdot MN\cdot 2\tilde{\sigma}_p^2\cdot\sqrt{d\log(4N^2M^2/\delta)} \\
&\leq \frac{2\eta\tilde{\sigma}_p^2 d}{N} \\
&= O(\eta\|\boldsymbol{\mu}\|_2^2)
\end{aligned}$$

where the second inequality is by Lemma C.4 and $-\ell_n^{\prime(t)} \leq 1$, the third inequality is by $d = \widetilde{\Omega}\big(\epsilon^{-2}N^2 d_h\big) \geq 4N\sqrt{\log(4N^2M^2/\delta)}$, the last inequality is by $N\cdot\mathrm{SNR}^2 = \Omega(1)$. Then by Definition B.1, we have

$$\begin{aligned}
|V_{n,i}^{(t)}| &= \Big|V_{n,i}^{(0)} + \sum_{s=0}^{t-1}(\rho_{V,n,i}^{(s+1)} - \rho_{V,n,i}^{(s)})\|\boldsymbol{w}_O\|_2^2\Big| \\
&\leq |V_{n,i}^{(0)}| + \sum_{s=0}^{t-1}|\rho_{V,n,i}^{(s+1)} - \rho_{V,n,i}^{(s)}| \cdot \|\boldsymbol{w}_O\|_2^2 \\
&\leq d_h^{-\frac{1}{4}} + O(\eta\|\boldsymbol{\mu}\|_2^2)\cdot\|\boldsymbol{w}_O\|_2^2\cdot O(\frac{1}{\eta d_h^{\frac{1}{4}}\|\boldsymbol{\mu}\|_2^2\|\boldsymbol{w}_O\|_2^2}) \\
&= O(d^{-\frac{1}{4}}),
\end{aligned}$$

where the first inequality is by triangle inequality, the second inequality is by Lemma C.2, which completes the proof.

**Lemma D.4** (Inner Products Hold Magnitude). *Let $T_0 = O(\frac{1}{\eta d_h^{\frac{1}{4}} \|\boldsymbol{\mu}\|_2^2 \|\boldsymbol{w}_O\|_2^2})$. Then under the same conditions as Theorem 4.1, we have*

$$|\langle \boldsymbol{q}_\pm^{(t)}, \boldsymbol{k}_\pm^{(t)}\rangle|, |\langle \boldsymbol{q}_{n,i}^{(t)}, \boldsymbol{k}_\pm^{(t)}\rangle|, |\langle \boldsymbol{q}_\pm^{(t)}, \boldsymbol{k}_{n,j}^{(t)}\rangle|, |\langle \boldsymbol{q}_{n,i}^{(t)}, \boldsymbol{k}_{n',j}^{(t)}\rangle|$$
$$= O\Big( \max\{\|\boldsymbol{\mu}\|_2^2, \sigma_p^2 d\} \cdot \sigma_h^2 \cdot \sqrt{d_h \log(6N^2 M^2/\delta)}\Big),$$

$$|\langle \boldsymbol{q}_\pm^{(t)}, \boldsymbol{q}_\mp^{(t)}\rangle|, |\langle \boldsymbol{q}_{n,i}^{(t)}, \boldsymbol{q}_\pm^{(t)}\rangle|, |\langle \boldsymbol{q}_{n,i}^{(t)}, \boldsymbol{q}_{n',j}^{(t)}\rangle|$$
$$= O\Big( \max\{\|\boldsymbol{\mu}\|_2^2, \sigma_p^2 d\} \cdot \sigma_h^2 \cdot \sqrt{d_h \log(6N^2 M^2/\delta)}\Big),$$

$$|\langle \boldsymbol{k}_\pm^{(t)}, \boldsymbol{k}_\mp^{(t)}\rangle|, |\langle \boldsymbol{k}_{n,i}^{(t)}, \boldsymbol{k}_\pm^{(t)}\rangle|, |\langle \boldsymbol{k}_{n,i}^{(t)}, \boldsymbol{k}_{n',j}^{(t)}\rangle|$$
$$= O\Big( \max\{\|\boldsymbol{\mu}\|_2^2, \sigma_p^2 d\} \cdot \sigma_h^2 \cdot \sqrt{d_h \log(6N^2 M^2/\delta)}\Big),$$

$$\|\boldsymbol{q}_\pm^{(t)}\|_2^2, \|\boldsymbol{k}_\pm^{(t)}\|_2^2 = \Theta(\|\boldsymbol{\mu}\|_2^2 \sigma_h^2 d_h),$$

$$\|\boldsymbol{q}_{n,i}^{(t)}\|_2^2, \|\boldsymbol{k}_{n,i}^{(t)}\|_2^2 = \Theta(\sigma_p^2 \sigma_h^2 d d_h)$$

*for $i, j \in [M]\backslash\{1\}$, $n, n' \in [N]$ and $t \in [0, T_0]$.*

The proof for Lemma D.4 is in Section F.3. Note that $\sigma_h^2 \leq \min\{\|\boldsymbol{\mu}\|_2^{-2}, (\sigma_p^2 d)^{-1}\} \cdot d_h^{-\frac{1}{2}} \cdot \big( \log(6N^2 M^2/\delta)\big)^{-\frac{3}{2}}$, thus $O\Big( \max\{\|\boldsymbol{\mu}\|_2^2, \sigma_p^2 d\} \cdot \sigma_h^2 \cdot \sqrt{d_h \log(6N^2 M^2/\delta)}\Big) = o(1)$.

**Lemma D.5** (V's Beginning of Learning Signals). *Under the same conditions as Theorem 4.1, there exist $T_1 = \frac{10M(3M+1)N}{\eta d_h^{\frac{1}{4}} (N\|\boldsymbol{\mu}\|_2^2 - 60M^2 C_p^2 \sigma_p^2 d)\|\boldsymbol{w}_O\|_2^2}$ such that the first element of vector $\boldsymbol{X}_n \boldsymbol{W}_V^{(t)} \boldsymbol{w}_O$ dominate its other elements, that is, $V_+^{(t)} \geq 3M \cdot |V_{n,i}^{(t)}|$ for all $n \in S_+$, $i \in [M]\backslash\{1\}$ and $V_-^{(t)} \leq -3M \cdot |V_{n,i}^{(t)}|$ for all $n \in S_-$, $i \in [M]\backslash\{1\}$.*

*Proof of Lemma D.5.* Let $C$ be a constant larger than $10M(3M+1)$, then as long as $N \cdot \text{SNR}^2 = \frac{N\|\boldsymbol{\mu}\|_2^2}{\sigma_p^2 d} \geq \frac{60CM^2 C_p^2}{C - 10M(3M+1)}$, we have $N\|\boldsymbol{\mu}\|_2^2 - 60M^2 C_p^2 \sigma_p^2 d \geq \frac{10M(3M+1)N\|\boldsymbol{\mu}\|_2^2}{C}$, and further get

$$T_1 = \frac{10M(3M+1)N}{\eta d_h^{\frac{1}{4}} (N\|\boldsymbol{\mu}\|_2^2 - 60M^2 C_p^2 \sigma_p^2 d)\|\boldsymbol{w}_O\|_2^2} \leq \frac{C}{\eta d_h^{\frac{1}{4}} \|\boldsymbol{\mu}\|_2^2 \|\boldsymbol{w}_O\|_2^2} = O\Big(\frac{1}{\eta d_h^{\frac{1}{4}} \|\boldsymbol{\mu}\|_2^2 \|\boldsymbol{w}_O\|_2^2}\Big),$$

which satisfies the time condition in Lemma D.3 and Lemma D.4. Then by Lemma D.1 and Lemma D.2 we have

$$-\ell_n'^{(t)} = \frac{1}{2} \pm o(1),$$

$$\frac{1}{M} - o(1) \leq softmax(\langle \boldsymbol{q}_\pm^{(t)}, \boldsymbol{k}_\pm^{(t)}\rangle) \leq \frac{1}{M} + o(1),$$

$$\frac{1}{M} - o(1) \leq softmax(\langle \boldsymbol{q}_{n,i}^{(t)}, \boldsymbol{k}_\pm^{(t)}\rangle) \leq \frac{1}{M} + o(1),$$

$$\frac{1}{M} - o(1) \leq softmax(\langle \boldsymbol{q}_\pm^{(t)}, \boldsymbol{k}_{n,j}^{(t)}\rangle) \leq \frac{1}{M} + o(1),$$

$$\frac{1}{M} - o(1) \leq softmax(\langle \boldsymbol{q}_{n,i}^{(t)}, \boldsymbol{k}_{n,j}^{(t)}\rangle) \leq \frac{1}{M} + o(1)$$

for $i, j \in [M] \backslash \{1\}$, $n \in [N]$ and $t \in [0, T_1]$. Plugging them in the update rule for $\gamma_{V,+}^{(t)}$ showed in Lemma B.2 and we have

$$
\begin{aligned}
&\gamma_{V,+}^{(t+1)} - \gamma_{V,+}^{(t)} \\
&= -\frac{\eta \|\boldsymbol{\mu}\|_2^2}{NM} \sum_{n \in S_+} \ell_n'^{(t)} \Big( \frac{\exp(\langle \boldsymbol{q}_+^{(t)}, \boldsymbol{k}_+^{(t)} \rangle)}{\exp(\langle \boldsymbol{q}_+^{(t)}, \boldsymbol{k}_+^{(t)} \rangle) + \sum\limits_{k=2}^{M} \exp(\langle \boldsymbol{q}_+^{(t)}, \boldsymbol{k}_{n,k}^{(t)} \rangle)} \\
&\quad + \sum_{j=2}^{M} \frac{\exp(\langle \boldsymbol{q}_{n,j}^{(t)}, \boldsymbol{k}_+^{(t)} \rangle)}{\exp(\langle \boldsymbol{q}_{n,j}^{(t)}, \boldsymbol{k}_+^{(t)} \rangle) + \sum\limits_{k=2}^{M} \exp(\langle \boldsymbol{q}_{n,j}^{(t)}, \boldsymbol{k}_{n,k}^{(t)} \rangle)} \Big) \qquad (18) \\
&\geq \frac{\eta \|\boldsymbol{\mu}\|_2^2}{NM} \cdot \frac{N}{4} \cdot \Big( \frac{1}{2} \pm o(1) \Big) \cdot M \Big( \frac{1}{M} \pm o(1) \Big) \\
&\geq \frac{\eta \|\boldsymbol{\mu}\|_2^2}{10M}.
\end{aligned}
$$

Then by Definition B.1 and taking a summation, we have

$$
\begin{aligned}
V_+^{(T_1)} &\geq -|V_+^{(0)}| + T_1 \frac{\eta \|\boldsymbol{\mu}\|_2^2}{10M} \|\boldsymbol{w}_O\|_2^2 \\
&= -d_h^{-\frac{1}{4}} + \frac{10M(3M+1)N}{\eta d_h^{\frac{1}{4}} (N\|\boldsymbol{\mu}\|_2^2 - 60M^2 C_p^2 \sigma_p^2 d) \|\boldsymbol{w}_O\|_2^2} \frac{\eta \|\boldsymbol{\mu}\|_2^2}{10M} \|\boldsymbol{w}_O\|_2^2 \qquad (19) \\
&= -d_h^{-\frac{1}{4}} + \frac{(3M+1)N\|\boldsymbol{\mu}\|_2^2}{d_h^{\frac{1}{4}} (N\|\boldsymbol{\mu}\|_2^2 - 60M^2 C_p^2 \sigma_p^2 d)}.
\end{aligned}
$$

Similarly, we have

$$
V_-^{(T_1)} \leq d_h^{-\frac{1}{4}} - \frac{(3M+1)N\|\boldsymbol{\mu}\|_2^2}{d_h^{\frac{1}{4}} (N\|\boldsymbol{\mu}\|_2^2 - 60M^2 C_p^2 \sigma_p^2 d)}. \qquad (20)
$$

Similarly, by $|\rho_{V,n,i}^{(t+1)} - \rho_{V,n,i}^{(t)}| \leq \frac{2\eta \tilde{\sigma}_p^2 d}{N}$ in (17), we have

$$
\begin{aligned}
|V_{n,i}^{(T_1)}| &\leq |V_{n,i}^{(0)}| + T_1 \frac{2\eta \tilde{\sigma}_p^2 d}{N} \|\boldsymbol{w}_O\|_2^2 \\
&= d_h^{-\frac{1}{4}} + \frac{10M(3M+1)N}{\eta d_h^{\frac{1}{4}} (N\|\boldsymbol{\mu}\|_2^2 - 60M^2 C_p^2 \sigma_p^2 d) \|\boldsymbol{w}_O\|_2^2} \frac{2\eta \tilde{\sigma}_p^2 d}{N} \|\boldsymbol{w}_O\|_2^2 \qquad (21) \\
&= d_h^{-\frac{1}{4}} + \frac{20M C_p^2 (3M+1) \sigma_p^2 d}{d_h^{\frac{1}{4}} (N\|\boldsymbol{\mu}\|_2^2 - 60M^2 C_p^2 \sigma_p^2 d)}.
\end{aligned}
$$

According to (19), (20) and (21), it is easy to verify that $V_+^{(T_1)} - 3M \cdot |V_{n,i}^{(T_1)}| \geq 0$ and $V_-^{(T_1)} + 3M \cdot |V_{n,i}^{(T_1)}| \leq 0$, which completes the proof.

### D.2  Stage II

In stage II, $\langle \boldsymbol{q}_+, \boldsymbol{k}_+ \rangle$, $\langle \boldsymbol{q}_{n,i}, \boldsymbol{k}_+ \rangle$ grows while $\langle \boldsymbol{q}_+, \boldsymbol{k}_{n,j} \rangle$, $\langle \boldsymbol{q}_{n,i}, \boldsymbol{k}_{n,j} \rangle$ decreases, resulting in attention focusing more and more on the signals and less on the noises. By the results of stage I, we have the following conditions at the beginning of stage II

$$
\begin{aligned}
V_+^{(T_1)} &\geq 3M \cdot |V_{n,i}^{(T_1)}|, \\
V_-^{(T_1)} &\leq -3M \cdot |V_{n,i}^{(T_1)}|, \\
|V_+^{(T_1)}|, |V_-^{(T_1)}|, |V_{n,i}^{(T_1)}| &= O(d_h^{-\frac{1}{4}}),
\end{aligned}
$$

$$|\langle \boldsymbol{q}_\pm^{(T_1)}, \boldsymbol{k}_\pm^{(T_1)}\rangle|, |\langle \boldsymbol{q}_{n,i}^{(T_1)}, \boldsymbol{k}_\pm^{(T_1)}\rangle|, |\langle \boldsymbol{q}_\pm^{(T_1)}, \boldsymbol{k}_{n,j}^{(T_1)}\rangle|, |\langle \boldsymbol{q}_{n,i}^{(T_1)}, \boldsymbol{k}_{n',j}^{(T_1)}\rangle|$$
$$= O\left( \max\{\|\boldsymbol{\mu}\|_2^2, \sigma_p^2 d\} \cdot \sigma_h^2 \cdot \sqrt{d_h \log(6N^2 M^2/\delta)}\right),$$

$$|\langle \boldsymbol{q}_\pm^{(T_1)}, \boldsymbol{q}_\mp^{(T_1)}\rangle|, |\langle \boldsymbol{q}_{n,i}^{(T_1)}, \boldsymbol{q}_\pm^{(T_1)}\rangle|, |\langle \boldsymbol{q}_{n,i}^{(T_1)}, \boldsymbol{q}_{n',j}^{(T_1)}\rangle|$$
$$= O\left( \max\{\|\boldsymbol{\mu}\|_2^2, \sigma_p^2 d\} \cdot \sigma_h^2 \cdot \sqrt{d_h \log(6N^2 M^2/\delta)}\right),$$

$$|\langle \boldsymbol{k}_\pm^{(T_1)}, \boldsymbol{k}_\mp^{(T_1)}\rangle|, |\langle \boldsymbol{k}_{n,i}^{(T_1)}, \boldsymbol{k}_\pm^{(T_1)}\rangle|, |\langle \boldsymbol{k}_{n,i}^{(T_1)}, \boldsymbol{k}_{n',j}^{(T_1)}\rangle|$$
$$= O\left( \max\{\|\boldsymbol{\mu}\|_2^2, \sigma_p^2 d\} \cdot \sigma_h^2 \cdot \sqrt{d_h \log(6N^2 M^2/\delta)}\right),$$

$$\|\boldsymbol{q}_\pm^{(T_1)}\|_2^2, \|\boldsymbol{k}_\pm^{(T_1)}\|_2^2 = \Theta(\|\boldsymbol{\mu}\|_2^2 \sigma_h^2 d_h),$$
$$\|\boldsymbol{q}_{n,i}^{(T_1)}\|_2^2, \|\boldsymbol{k}_{n,i}^{(T_1)}\|_2^2 = \Theta(\sigma_p^2 \sigma_h^2 d d_h)$$

for $i, j \in [M]\setminus\{1\}, n, n' \in [N]$.

Some of the proofs at this stage are based on the above conditions.

**Notations.** To better characterize the gap between different inner products, we define the following notations:

- denote $\Lambda_{n,+,j}^{(t)} = \langle \boldsymbol{q}_+^{(t)}, \boldsymbol{k}_+^{(t)}\rangle - \langle \boldsymbol{q}_+^{(t)}, \boldsymbol{k}_{n,j}^{(t)}\rangle, \quad n \in S_+$.
- denote $\Lambda_{n,-,j}^{(t)} = \langle \boldsymbol{q}_-^{(t)}, \boldsymbol{k}_-^{(t)}\rangle - \langle \boldsymbol{q}_-^{(t)}, \boldsymbol{k}_{n,j}^{(t)}\rangle, \quad n \in S_-$.
- denote $\Lambda_{n,i,+,j}^{(t)} = \langle \boldsymbol{q}_{n,i}^{(t)}, \boldsymbol{k}_+^{(t)}\rangle - \langle \boldsymbol{q}_{n,i}^{(t)}, \boldsymbol{k}_{n,j}^{(t)}\rangle, \quad n \in S_+$.
- denote $\Lambda_{n,i,-,j}^{(t)} = \langle \boldsymbol{q}_{n,i}^{(t)}, \boldsymbol{k}_-^{(t)}\rangle - \langle \boldsymbol{q}_{n,i}^{(t)}, \boldsymbol{k}_{n,j}^{(t)}\rangle, \quad n \in S_-$.

**Lemma D.6** (Upper bound of V). *Let $T_0 = O\left(\frac{1}{\eta \|\boldsymbol{\mu}\|_2^2 \|\boldsymbol{w}_O\|_2^2 \log(6N^2 M^2/\delta)}\right)$. Then under the same conditions as Theorem 4.1, we have*

$$|V_+^{(t)}|, |V_-^{(t)}|, |V_{n,i}^{(t)}| = o(1)$$

*for $t \in [0, T_0]$.*

The proof of Lemma D.6 is similar to that of Lemma D.3, except that the time $T_0$ is changed.

Let $T_2 = \Theta\left(\frac{1}{\eta \|\boldsymbol{\mu}\|_2^2 \|\boldsymbol{w}_O\|_2^2 \log(6N^2 M^2/\delta)}\right)$, then by Lemma D.6 and Lemma D.1 we have $\frac{1}{2} - o(1) \le -\ell_n'^{(t)} \le \frac{1}{2} + o(1)$ for $n \in [N], t \in [T_1, T_2]$, which can simplify the calculations of $\alpha$ and $\beta$ defined in Definition B.3 by replacing $-\ell_n'^{(t)}$ by their bounds. Next we prove the following four propositions $\mathcal{B}(t), \mathcal{C}(t), \mathcal{D}(t), \mathcal{E}(t)$ by induction on t for $t \in [T_1, T_2]$:

- $\mathcal{B}(t)$ :
$$V_+^{(t)} \ge \eta C_3 \|\boldsymbol{\mu}\|_2^2 \|\boldsymbol{w}_O\|_2^2 (t - T_1)$$
$$V_-^{(t)} \le -\eta C_3 \|\boldsymbol{\mu}\|_2^2 \|\boldsymbol{w}_O\|_2^2 (t - T_1)$$
$$V_+^{(t)} \ge 3M \cdot |V_{n,i}^{(t)}|,$$
$$V_-^{(t)} \le -3M \cdot |V_{n,i}^{(t)}|,$$
$$|V_\pm^{(t)}| \le O(d_h^{-\frac{1}{4}}) + \eta C_4 \|\boldsymbol{\mu}\|_2^2 \|\boldsymbol{w}_O\|_2^2 (t - T_1)$$
$$|V_{n,i}^{(t)}| \le O(d_h^{-\frac{1}{4}}) + \eta C_4 \|\boldsymbol{\mu}\|_2^2 \|\boldsymbol{w}_O\|_2^2 (t - T_1)$$

for $i \in [M]\setminus\{1\}, n \in [N]$.

- $\mathcal{C}(t)$ :

$$\|\boldsymbol{q}_{\pm}^{(t)}\|_2^2, \|\boldsymbol{k}_{\pm}^{(t)}\|_2^2 = \Theta\Big(\|\boldsymbol{\mu}\|_2^2 \sigma_h^2 d_h\Big),$$

$$\|\boldsymbol{q}_{n,i}^{(t)}\|_2^2, \|\boldsymbol{k}_{n,i}^{(t)}\|_2^2 = \Theta\Big(\sigma_p^2 \sigma_h^2 d d_h\Big),$$

$$|\langle \boldsymbol{q}_{+}^{(t)}, \boldsymbol{q}_{-}^{(t)}\rangle|, |\langle \boldsymbol{q}_{\pm}^{(t)}, \boldsymbol{q}_{n,i}^{(t)}\rangle|, |\langle \boldsymbol{q}_{n,i}^{(t)}, \boldsymbol{q}_{n',j}^{(t)}\rangle| = o(1),$$

$$|\langle \boldsymbol{k}_{+}^{(t)}, \boldsymbol{k}_{-}^{(t)}\rangle|, |\langle \boldsymbol{k}_{\pm}^{(t)}, \boldsymbol{k}_{n,i}^{(t)}\rangle|, |\langle \boldsymbol{k}_{n,i}^{(t)}, \boldsymbol{k}_{n',j}^{(t)}\rangle| = o(1),$$

for $i, j \in [M]\setminus\{1\}, n, n' \in [N], i \neq j$ or $n \neq n'$.

- $\mathcal{D}(t)$ :

$$\langle \boldsymbol{q}_{\pm}^{(t+1)}, \boldsymbol{k}_{\pm}^{(t+1)}\rangle \geq \langle \boldsymbol{q}_{\pm}^{(t)}, \boldsymbol{k}_{\pm}^{(t)}\rangle$$

$$\langle \boldsymbol{q}_{n,i}^{(t+1)}, \boldsymbol{k}_{\pm}^{(t+1)}\rangle \geq \langle \boldsymbol{q}_{n,i}^{(t)}, \boldsymbol{k}_{\pm}^{(t)}\rangle$$

$$\langle \boldsymbol{q}_{\pm}^{(t+1)}, \boldsymbol{k}_{n,j}^{(t+1)}\rangle \leq \langle \boldsymbol{q}_{\pm}^{(t)}, \boldsymbol{k}_{n,j}^{(t)}\rangle$$

$$\langle \boldsymbol{q}_{n,i}^{(t+1)}, \boldsymbol{k}_{n,j}^{(t+1)}\rangle \leq \langle \boldsymbol{q}_{n,i}^{(t)}, \boldsymbol{k}_{n,j}^{(t)}\rangle$$

$$\Lambda_{n,\pm,j}^{(t+1)} \geq \log\Big(\exp(\Lambda_{n,\pm,j}^{(T_1)}) + \frac{\eta^2 C_8 \|\boldsymbol{\mu}\|_2^4 \|\boldsymbol{w}_O\|_2^2 d_h^{\frac{1}{2}}}{N\big(\log(6N^2 M^2/\delta)\big)^2} \cdot (t - T_1)(t - T_1 + 1)\Big)$$

$$\Lambda_{n,i,\pm,j}^{(t+1)} \geq \log\Big(\exp(\Lambda_{n,i,\pm,j}^{(T_1)}) + \frac{\eta^2 C_8 \sigma_p^2 d \|\boldsymbol{\mu}\|_2^2 \|\boldsymbol{w}_O\|_2^2 d_h^{\frac{1}{2}}}{N\big(\log(6N^2 M^2/\delta)\big)^2} \cdot (t - T_1)(t - T_1 + 1)\Big)$$

for $i, j \in [M]\setminus\{1\}, n \in [N]$.

- $\mathcal{E}(t)$ :

$$|\langle \boldsymbol{q}_{\pm}^{(t)}, \boldsymbol{k}_{\pm}^{(t)}\rangle|, |\langle \boldsymbol{q}_{\pm}^{(t)}, \boldsymbol{k}_{n,j}^{(t)}\rangle|, |\langle \boldsymbol{q}_{n,i}^{(t)}, \boldsymbol{k}_{\pm}^{(t)}\rangle|, |\langle \boldsymbol{q}_{n,i}^{(t)}, \boldsymbol{k}_{n,j}^{(t)}\rangle| \leq \log(d_h^{\frac{1}{2}})$$

$$|\langle \boldsymbol{q}_{\pm}^{(t)}, \boldsymbol{k}_{\mp}^{(t)}\rangle|, |\langle \boldsymbol{q}_{n,i}^{(t)}, \boldsymbol{k}_{\overline{n},j}^{(t)}\rangle| = o(1)$$

for $i, j \in [M]\setminus\{1\}, n, \overline{n} \in [N], n \neq \overline{n}$.

By the results of Stage I, we know that $\mathcal{B}(T_1), \mathcal{C}(T_1), \mathcal{E}(T_1)$ are true. To prove that $\mathcal{B}(t), \mathcal{C}(t), \mathcal{D}(t)$ and $\mathcal{E}(t)$ are true in stage 2, we will prove the following claims holds for $t \in [T_1, T_2]$:

**Claim 1.** $\mathcal{D}(T_1), \ldots, \mathcal{D}(t-1) \Longrightarrow \mathcal{B}(t+1)$

**Claim 2.** $\mathcal{B}(T_1), \ldots, \mathcal{B}(t), \mathcal{C}(T_1), \ldots, \mathcal{C}(t), \mathcal{D}(T_1), \ldots, \mathcal{D}(t-1) \Longrightarrow \mathcal{D}(t)$

**Claim 3.** $\mathcal{B}(T_1), \ldots, \mathcal{B}(t), \mathcal{D}(T_1), \ldots, \mathcal{D}(t-1), \mathcal{E}(T_1), \ldots, \mathcal{E}(t) \Longrightarrow \mathcal{C}(t+1)$

**Claim 4.** $\mathcal{B}(T_1), \ldots, \mathcal{B}(t), \mathcal{C}(T_1), \ldots, \mathcal{C}(t), \mathcal{D}(T_1), \ldots, \mathcal{D}(t-1) \Longrightarrow \mathcal{E}(t+1)$

### D.2.1 Proof of Claim 1

By the results of Stage I, we have

$$|\langle \boldsymbol{q}_{\pm}^{(T_1)}, \boldsymbol{k}_{\pm}^{(T_1)}\rangle|, |\langle \boldsymbol{q}_{\pm}^{(T_1)}, \boldsymbol{k}_{n,j}^{(T_1)}\rangle|, |\langle \boldsymbol{q}_{n,i}^{(T_1)}, \boldsymbol{k}_{\pm}^{(T_1)}\rangle|, |\langle \boldsymbol{q}_{n,i}^{(T_1)}, \boldsymbol{k}_{n,j}^{(T_1)}\rangle| = o(1)$$

Assume that $\mathcal{D}(T_1), \ldots, \mathcal{D}(t-1)$ $(t \in [T_1, T_2])$ are true, then $\langle \boldsymbol{q}_{\pm}^{(s)}, \boldsymbol{k}_{\pm}^{(s)}\rangle$, $\langle \boldsymbol{q}_{n,i}^{(s)}, \boldsymbol{k}_{\pm}^{(s)}\rangle$ are monotonically non-decreasing and $\langle \boldsymbol{q}_{\pm}^{(s)}, \boldsymbol{k}_{n,j}^{(s)}\rangle$, $\langle \boldsymbol{q}_{n,i}^{(s)}, \boldsymbol{k}_{n,j}^{(s)}\rangle$ are monotonically non-increasing for $s \in [T_1, t-1]$, so we have

$$\langle \boldsymbol{q}_{\pm}^{(s)}, \boldsymbol{k}_{\pm}^{(s)}\rangle, \langle \boldsymbol{q}_{n,i}^{(s)}, \boldsymbol{k}_{\pm}^{(s)}\rangle \geq -o(1),$$

$$\langle \boldsymbol{q}_{\pm}^{(s)}, \boldsymbol{k}_{n,j}^{(s)}\rangle, \langle \boldsymbol{q}_{n,i}^{(s)}, \boldsymbol{k}_{n,j}^{(s)}\rangle \leq o(1),$$

for $s \in [T_1, t]$. Further we have the lower bounds for the attention on signal $\boldsymbol{\mu}_\pm$ as follows for $s \in [T_1, t]$:

$$
\begin{aligned}
softmax(\langle \boldsymbol{q}_\pm^{(s)}, \boldsymbol{k}_\pm^{(s)} \rangle) &\geq \frac{\exp(-o(1))}{\exp(-o(1)) + (M-1)\exp(o(1))} \\
&= \frac{1}{1 + (M-1)\exp(o(1))} \\
&= \frac{1}{1 + (M-1) + (M-1)o(1)} \\
&= \frac{1}{M} - o(1),
\end{aligned}
\tag{22}
$$

where the second equality is by $\exp(o(1)) = 1 + o(1)$. Similarly, we have $softmax(\langle \boldsymbol{q}_{n,i}^{(s)}, \boldsymbol{k}_\pm^{(s)} \rangle) \geq \frac{1}{M} - o(1)$. Plugging them in the update rule for $\gamma_{V,+}^{(s)}$ showed in Lemma B.2 and we have

$$
\begin{aligned}
&\gamma_{V,+}^{(s+1)} - \gamma_{V,+}^{(s)} \\
&= -\frac{\eta\|\boldsymbol{\mu}\|_2^2}{NM} \sum_{n \in S_+} \ell_n'^{(t)} \Big( \frac{\exp(\langle \boldsymbol{q}_+^{(s)}, \boldsymbol{k}_+^{(s)} \rangle)}{\exp(\langle \boldsymbol{q}_+^{(s)}, \boldsymbol{k}_+^{(s)} \rangle) + \sum\limits_{k=2}^{M} \exp(\langle \boldsymbol{q}_+^{(s)}, \boldsymbol{k}_{n,k}^{(s)} \rangle)} \\
&\quad + \sum_{j=2}^{M} \frac{\exp(\langle \boldsymbol{q}_{n,j}^{(s)}, \boldsymbol{k}_+^{(s)} \rangle)}{\exp(\langle \boldsymbol{q}_{n,j}^{(s)}, \boldsymbol{k}_+^{(s)} \rangle) + \sum\limits_{k=2}^{M} \exp(\langle \boldsymbol{q}_{n,j}^{(s)}, \boldsymbol{k}_{n,k}^{(s)} \rangle)} \Big) \\
&= \frac{\eta\|\boldsymbol{\mu}\|_2^2}{NM} \cdot \frac{N}{4} \cdot \Big(\frac{1}{2} - o(1)\Big) \cdot M\Big(\frac{1}{M} - o(1)\Big) \\
&\geq \frac{\eta\|\boldsymbol{\mu}\|_2^2}{10M},
\end{aligned}
\tag{23}
$$

for $s \in [T_1, t]$. Then by Definition B.1 and taking a summation, we have

$$
\begin{aligned}
V_+^{(t+1)} &\geq V_+^{(T_1)} + (t - T_1 + 1)\frac{\eta\|\boldsymbol{\mu}\|_2^2}{10M}\|\boldsymbol{w}_O\|_2^2 \\
&\geq \eta C_3\|\boldsymbol{\mu}\|_2^2\|\boldsymbol{w}_O\|_2^2(t - T_1 + 1),
\end{aligned}
\tag{24}
$$

where the last inequality is by $V_+^{(T_1)} \geq 0$ and $M = \Theta(1)$. Similarly, we have

$$
V_-^{(t+1)} \leq -\eta C_3\|\boldsymbol{\mu}\|_2^2\|\boldsymbol{w}_O\|_2^2(t - T_1 + 1).
$$

By $|\rho_{V,n,i}^{(t+1)} - \rho_{V,n,i}^{(t)}| \leq \frac{2\eta\tilde{\sigma}_p^2 d}{N}$ in (17) and taking a summation, we have

$$
|V_{n,i}^{(t+1)}| \leq |V_{n,i}^{(T_1)}| + (t - T_1 + 1)\frac{2\eta\tilde{\sigma}_p^2 d}{N}\|\boldsymbol{w}_O\|_2^2 \leq |V_{n,i}^{(T_1)}| + (t - T_1 + 1)\frac{2\eta C_p^2 \sigma_p^2 d}{N}\|\boldsymbol{w}_O\|_2^2.
\tag{25}
$$

Combining (24) and (25) we have

$$
\begin{aligned}
&V_+^{(t+1)} - 3M \cdot |V_{n,i}^{(t+1)}| \\
&\geq V_+^{(T_1)} + (t - T_1 + 1)\frac{\eta\|\boldsymbol{\mu}\|_2^2}{10M}\|\boldsymbol{w}_O\|_2^2 - 3M \cdot \Big(|V_{n,i}^{(T_1)}| + (t - T_1 + 1)\frac{2\eta C_p^2 \sigma_p^2 d}{N}\|\boldsymbol{w}_O\|_2^2\Big) \\
&\geq V_+^{(T_1)} - 3M \cdot |V_{n,i}^{(T_1)}| + (t - T_1 + 1)\frac{\eta\big(N\|\boldsymbol{\mu}\|_2^2 - 60M^2 C_p^2 \sigma_p^2 d\big)}{10NM} \\
&\geq 0,
\end{aligned}
\tag{26}
$$

where the last inequality is by $V_+^{(T_1)} \geq 3M \cdot |V_{n,i}^{(T_1)}|$ and requires $N \cdot \text{SNR}^2 \geq 60M^2 C_p^2$. The proof for $V_-^{(t+1)} \leq -3M \cdot |V_{n,i}^{(t+1)}|$ is the same.

Next, we prove the upper bound for $V_\pm$ and $V_{n,i}$. Based on the upper bound of attention($<1$) and $-\ell'_n \leq 1$, we have

$$\gamma_{V,+}^{(s+1)} \leq \gamma_{V,+}^{(s)} - \frac{\eta}{NM} \sum_{n \in S_+} \ell'_n(\theta(s))(\|\boldsymbol{\mu}\|_2^2 + \sum_{j=2}^M \|\boldsymbol{\mu}\|_2^2)$$
$$\leq \gamma_{V,+}^{(s)} + \frac{3\eta\|\boldsymbol{\mu}\|_2^2}{4} \tag{27}$$
$$\leq \gamma_{V,+}^{(s)} + \eta C_4 \|\boldsymbol{\mu}\|_2^2$$

Then we can get that

$$|V_+^{(t+1)}| \leq V_+^{(T_1)} + (\gamma_{V,+}^{(t+1)} - \gamma_{V,+}^{(T_1)})\|\boldsymbol{w}_O\|_2^2$$
$$\leq V_+^{(T_1)} + \sum_{s=T_1}^t \eta C_4 \|\boldsymbol{\mu}\|_2^2 \|\boldsymbol{w}_O\|_2^2 \tag{28}$$
$$\leq O(d_h^{-\frac{1}{4}}) + \eta C_4 \|\boldsymbol{\mu}\|_2^2 \|\boldsymbol{w}_O\|_2^2(t - T_1 + 1)$$

where the first inequality is by the monotonicity of $\gamma_{V,+}$ and the definition of $V_+$, the last inequality is by the result of stage 1 where $V_+^{(T_1)} = O(d^{-1})$. Similarly, we have

$$|V_-^{(t+1)}| \leq O(d_h^{-\frac{1}{4}}) + \eta C_4 \|\boldsymbol{\mu}\|_2^2 \|\boldsymbol{w}_O\|_2^2(t - T_1 + 1) \tag{29}$$

which completes the proof for the upper bound of $V_\pm$.

Expanding (25) yields

$$|V_{n,i}^{(t+1)}| \leq |V_{n,i}^{(T_1)}| + \frac{2\eta C_p^2 \sigma_p^2 d}{N} \cdot \|\boldsymbol{w}_O\|_2^2(t - T_1 + 1)$$
$$\leq O(d_h^{-\frac{1}{4}}) + \eta C_4 \|\boldsymbol{\mu}\|_2^2 \|\boldsymbol{w}_O\|_2^2(t - T_1 + 1) \tag{30}$$

where the last inequality is by the result of phase 1 where $|V_{n,i}^{(T_1)}| = O(d_h^{-\frac{1}{4}})$ and the condition that $N \cdot \text{SNR}^2 > \Omega(1)$.

### D.2.2 Proof of Claim 2

By the results of F.5, we have the dynamic of $\langle \boldsymbol{q}, \boldsymbol{k} \rangle$ as follows

$$\langle \boldsymbol{q}_+^{(s+1)}, \boldsymbol{k}_+^{(s+1)} \rangle - \langle \boldsymbol{q}_+^{(s)}, \boldsymbol{k}_+^{(s)} \rangle$$
$$\geq \frac{\eta^2 C_6 \|\boldsymbol{\mu}\|_2^6 \|\boldsymbol{w}_O\|_2^2 \sigma_h^2 d_h(s - T_1)}{N} \cdot \frac{1}{\exp(\Lambda_{n,+,j}^{(s)})}, \tag{31}$$

$$\langle \boldsymbol{q}_-^{(s+1)}, \boldsymbol{k}_-^{(s+1)} \rangle - \langle \boldsymbol{q}_-^{(s)}, \boldsymbol{k}_-^{(s)} \rangle$$
$$\geq \frac{\eta^2 C_6 \|\boldsymbol{\mu}\|_2^6 \|\boldsymbol{w}_O\|_2^2 \sigma_h^2 d_h(s - T_1)}{N} \cdot \frac{1}{\exp(\Lambda_{n,-,j}^{(s)})}, \tag{32}$$

$$\langle \boldsymbol{q}_+^{(s+1)}, \boldsymbol{k}_{n,j}^{(s+1)} \rangle - \langle \boldsymbol{q}_+^{(s)}, \boldsymbol{k}_{n,j}^{(s)} \rangle$$
$$\leq -\frac{\eta^2 C_6 \sigma_p^2 d \|\boldsymbol{\mu}\|_2^4 \|\boldsymbol{w}_O\|_2^2 \sigma_h^2 d_h(s - T_1)}{N} \cdot \frac{1}{\exp(\Lambda_{n,+,j}^{(s)})}, \tag{33}$$

$$\langle \boldsymbol{q}_-^{(s+1)}, \boldsymbol{k}_{n,j}^{(s+1)} \rangle - \langle \boldsymbol{q}_-^{(s)}, \boldsymbol{k}_{n,j}^{(s)} \rangle$$
$$\leq -\frac{\eta^2 C_6 \sigma_p^2 d \|\boldsymbol{\mu}\|_2^4 \|\boldsymbol{w}_O\|_2^2 \sigma_h^2 d_h(s - T_1)}{N} \cdot \frac{1}{\exp(\Lambda_{n,-,j}^{(s)})}, \tag{34}$$

$$\langle \boldsymbol{q}_{n,i}^{(s+1)}, \boldsymbol{k}_+^{(s+1)}\rangle - \langle \boldsymbol{q}_{n,i}^{(s)}, \boldsymbol{k}_+^{(s)}\rangle$$

$$\geq \frac{\eta^2 C_6 \sigma_p^2 d \|\boldsymbol{\mu}\|_2^4 \|\boldsymbol{w}_O\|_2^2 \sigma_h^2 d_h (s-T_1)}{N} \cdot \frac{1}{\exp(\Lambda_{n,i,+,j}^{(s)})}, \tag{35}$$

$$\langle \boldsymbol{q}_{n,i}^{(s+1)}, \boldsymbol{k}_-^{(s+1)}\rangle - \langle \boldsymbol{q}_{n,i}^{(s)}, \boldsymbol{k}_-^{(s)}\rangle$$

$$\geq \frac{\eta^2 C_6 \sigma_p^2 d \|\boldsymbol{\mu}\|_2^4 \|\boldsymbol{w}_O\|_2^2 \sigma_h^2 d_h (s-T_1)}{N} \cdot \frac{1}{\exp(\Lambda_{n,i,-,j}^{(s)})}, \tag{36}$$

$$\langle \boldsymbol{q}_{n,i}^{(s+1)}, \boldsymbol{k}_{n,j}^{(s+1)}\rangle - \langle \boldsymbol{q}_{n,i}^{(s)}, \boldsymbol{k}_{n,j}^{(s)}\rangle$$

$$\leq -\frac{\eta^2 C_6 \sigma_p^4 d^2 \|\boldsymbol{\mu}\|_2^2 \|\boldsymbol{w}_O\|_2^2 \sigma_h^2 d_h (s-T_1)}{N} \cdot \frac{1}{\exp(\Lambda_{n,i,\pm,j}^{(s)})} \tag{37}$$

for $s \in [T_1, t]$. The seven equations above show that $\langle \boldsymbol{q}_\pm^{(s)}, \boldsymbol{k}_\pm^{(s)}\rangle$, $\langle \boldsymbol{q}_{n,i}^{(s)}, \boldsymbol{k}_\pm^{(s)}\rangle$ are monotonically increasing and $\langle \boldsymbol{q}_\pm^{(s)}, \boldsymbol{k}_{n,j}^{(s)}\rangle$, $\langle \boldsymbol{q}_{n,i}^{(s)}, \boldsymbol{k}_{n,j}^{(s)}\rangle$ are monotonically decreasing. Next, we provide the logarithmic increasing lower bounds of $\Lambda_{n,\pm,j}^{(s+1)}$ and $\Lambda_{n,i,\pm,j}^{(s+1)}$.

By (31) and (33), we have

$$\Lambda_{n,+,j}^{(s+1)} - \Lambda_{n,+,j}^{(s)} = (\langle \boldsymbol{q}_+^{(s+1)}, \boldsymbol{k}_+^{(s+1)}\rangle - \langle \boldsymbol{q}_+^{(s)}, \boldsymbol{k}_+^{(s)}\rangle) - (\langle \boldsymbol{q}_+^{(s+1)}, \boldsymbol{k}_{n,j}^{(s+1)}\rangle - \langle \boldsymbol{q}_+^{(s)}, \boldsymbol{k}_{n,j}^{(s)}\rangle)$$

$$\geq \frac{\eta^2 C_6 \|\boldsymbol{\mu}\|_2^6 \|\boldsymbol{w}_O\|_2^2 \sigma_h^2 d_h (s-T_1)}{N} \cdot \frac{1}{\exp(\Lambda_{n,+,j}^{(s)})}$$

$$+ \frac{\eta^2 C_6 \sigma_p^2 d \|\boldsymbol{\mu}\|_2^4 \|\boldsymbol{w}_O\|_2^2 \sigma_h^2 d_h (s-T_1)}{N} \cdot \frac{1}{\exp(\Lambda_{n,+,j}^{(s)})} \tag{38}$$

$$\geq \frac{\eta^2 C_7 \max\{\|\boldsymbol{\mu}\|_2^2, \sigma_p^2 d\} \|\boldsymbol{\mu}\|_2^4 \|\boldsymbol{w}_O\|_2^2 \sigma_h^2 d_h (s-T_1)}{N} \cdot \frac{1}{\exp(\Lambda_{n,+,j}^{(s)})}$$

$$\geq \frac{\eta^2 C_7 \|\boldsymbol{\mu}\|_2^4 \|\boldsymbol{w}_O\|_2^2 d_h^{\frac{1}{2}} (s-T_1)}{N \big(\log(6N^2 M^2/\delta)\big)^2} \cdot \frac{1}{\exp(\Lambda_{n,+,j}^{(s)})},$$

where the last inequality is by $\sigma_h^2 \geq \min\{\|\boldsymbol{\mu}\|_2^{-2}, (\sigma_p^2 d)^{-1}\} d_h^{-\frac{1}{2}} (\log(6N^2 M^2/\delta))^{-2}$. Multiply both sides simultaneously by $\exp(\Lambda_{n,+,j}^{(s)})$ and get

$$\exp(\Lambda_{n,+,j}^{(s)}) \Big(\Lambda_{n,+,j}^{(s+1)} - \Lambda_{n,+,j}^{(s)}\Big) \geq \frac{\eta^2 C_7 \|\boldsymbol{\mu}\|_2^4 \|\boldsymbol{w}_O\|_2^2 d_h^{\frac{1}{2}} (s-T_1)}{N \big(\log(6N^2 M^2/\delta)\big)^2}. \tag{39}$$

Taking a summation from $T_1$ to $t$ and get

$$\sum_{s=T_1}^{t} \exp(\Lambda_{n,+,j}^{(s)}) \Big(\Lambda_{n,+,j}^{(s+1)} - \Lambda_{n,+,j}^{(s)}\Big)$$

$$\geq \sum_{s=T_1}^{t} \frac{\eta^2 C_7 \|\boldsymbol{\mu}\|_2^4 \|\boldsymbol{w}_O\|_2^2 d_h^{\frac{1}{2}} (s-T_1)}{N \big(\log(6N^2 M^2/\delta)\big)^2} \tag{40}$$

$$\geq \frac{\eta^2 C_8 \|\boldsymbol{\mu}\|_2^4 \|\boldsymbol{w}_O\|_2^2 d_h^{\frac{1}{2}}}{N \big(\log(6N^2 M^2/\delta)\big)^2} \cdot (t-T_1)(t-T_1+1).$$

By the property that $\Lambda_{n,+,j}^{(t)}$ is monotonically increasing, we have

$$\int_{\Lambda_{n,+,j}^{(T_1)}}^{\Lambda_{n,+,j}^{(t+1)}} \exp(x) dx \geq \sum_{s=T_1}^{t} \exp(\Lambda_{n,+,j}^{(s)}) \Big(\Lambda_{n,+,j}^{(s+1)} - \Lambda_{n,+,j}^{(s)}\Big)$$

$$\geq \frac{\eta^2 C_8 \|\boldsymbol{\mu}\|_2^4 \|\boldsymbol{w}_O\|_2^2 d_h^{\frac{1}{2}}}{N \big(\log(6N^2 M^2/\delta)\big)^2} \cdot (t-T_1)(t-T_1+1). \tag{41}$$

By $\int_{\Lambda_{n,+,j}^{(T_1)}}^{\Lambda_{n,+,j}^{(t+1)}} \exp(x)dx = \exp(\Lambda_{n,+,j}^{(t+1)}) - \exp(\Lambda_{n,+,j}^{(T_1)})$ we get

$$\Lambda_{n,+,j}^{(t+1)} \geq \log\Big( \exp(\Lambda_{n,+,j}^{(T_1)}) + \frac{\eta^2 C_8 \|\boldsymbol{\mu}\|_2^4 \|\boldsymbol{w}_O\|_2^2 d_h^{\frac{1}{2}}}{N\big(\log(6N^2M^2/\delta)\big)^2} \cdot (t-T_1)(t-T_1+1)\Big). \qquad (42)$$

Similarly, we have

$$\Lambda_{n,-,j}^{(t+1)} \geq \log\Big( \exp(\Lambda_{n,-,j}^{(T_1)}) + \frac{\eta^2 C_8 \|\boldsymbol{\mu}\|_2^4 \|\boldsymbol{w}_O\|_2^2 d_h^{\frac{1}{2}}}{N\big(\log(6N^2M^2/\delta)\big)^2} \cdot (t-T_1)(t-T_1+1)\Big). \qquad (43)$$

By (35) and (37), we have

$$\Lambda_{n,i,+,j}^{(s+1)} - \Lambda_{n,i,+,j}^{(s)} = (\langle \boldsymbol{q}_{n,i}^{(s+1)}, \boldsymbol{k}_+^{(s+1)}\rangle - \langle \boldsymbol{q}_{n,i}^{(s)}, \boldsymbol{k}_+^{(s)}\rangle) - (\langle \boldsymbol{q}_{n,i}^{(s+1)}, \boldsymbol{k}_{n,j}^{(s+1)}\rangle - \langle \boldsymbol{q}_{n,i}^{(s)}, \boldsymbol{k}_{n,j}^{(s)}\rangle)$$

$$\geq \frac{\eta^2 C_6 \sigma_p^2 d \|\boldsymbol{\mu}\|_2^4 \|\boldsymbol{w}_O\|_2^2 \sigma_h^2 d_h (s-T_1)}{N} \cdot \frac{1}{\exp(\Lambda_{n,i,+,j}^{(s)})}$$

$$+ \frac{\eta^2 C_6 \sigma_p^4 d^2 \|\boldsymbol{\mu}\|_2^2 \|\boldsymbol{w}_O\|_2^2 \sigma_h^2 d_h (s-T_1)}{N} \cdot \frac{1}{\exp(\Lambda_{n,i,+,j}^{(s)})}$$

$$\geq \frac{\eta^2 C_7 \max\{\|\boldsymbol{\mu}\|_2^2, \sigma_p^2 d\} \sigma_p^2 d \|\boldsymbol{\mu}\|_2^2 \|\boldsymbol{w}_O\|_2^2 \sigma_h^2 d_h (s-T_1)}{N} \cdot \frac{1}{\exp(\Lambda_{n,i,+,j}^{(s)})}$$

$$\geq \frac{\eta^2 C_7 \sigma_p^2 d \|\boldsymbol{\mu}\|_2^2 \|\boldsymbol{w}_O\|_2^2 d_h^{\frac{1}{2}} (s-T_1)}{N\big(\log(6N^2M^2/\delta)\big)^2} \cdot \frac{1}{\exp(\Lambda_{n,i,+,j}^{(s)})}.$$

$$\qquad (44)$$

Then using the similar method as for $\Lambda_{n,\pm,j}^{(t)}$, we get

$$\Lambda_{n,i,\pm,j}^{(t+1)} \geq \log\Big( \exp(\Lambda_{n,i,\pm,j}^{(T_1)}) + \frac{\eta^2 C_8 \sigma_p^2 d \|\boldsymbol{\mu}\|_2^2 \|\boldsymbol{w}_O\|_2^2 d_h^{\frac{1}{2}}}{N\big(\log(6N^2M^2/\delta)\big)^2} \cdot (t-T_1)(t-T_1+1)\Big), \qquad (45)$$

which complete the proof. The proof for Claim 3 is in Section F.8

### D.2.3  Proof of Claim 4

By the results of F.6, we have

$$\langle \boldsymbol{q}_+^{(s+1)}, \boldsymbol{k}_+^{(s+1)}\rangle - \langle \boldsymbol{q}_+^{(s)}, \boldsymbol{k}_+^{(s)}\rangle \leq \frac{\eta C_{10} \|\boldsymbol{\mu}\|_2^4 \sigma_h^2 d_h}{\exp(\langle \boldsymbol{q}_+^{(s)}, \boldsymbol{k}_+^{(s)}\rangle)} \qquad (46)$$

for $s \in [T_1, t]$. Further we have

$$\exp(\langle \boldsymbol{q}_+^{(s+1)}, \boldsymbol{k}_+^{(s+1)}\rangle) \leq \exp\Big( \langle \boldsymbol{q}_+^{(s)}, \boldsymbol{k}_+^{(s)}\rangle + \frac{\eta C_{10} \|\boldsymbol{\mu}\|_2^4 \sigma_h^2 d_h}{\exp(\langle \boldsymbol{q}_+^{(s)}, \boldsymbol{k}_+^{(s)}\rangle)}\Big)$$

$$= \exp\Big( \langle \boldsymbol{q}_+^{(s)}, \boldsymbol{k}_+^{(s)}\rangle\Big) \cdot \exp\Big( \frac{\eta C_{10} \|\boldsymbol{\mu}\|_2^4 \sigma_h^2 d_h}{\exp(\langle \boldsymbol{q}_+^{(s)}, \boldsymbol{k}_+^{(s)}\rangle)}\Big) \qquad (47)$$

$$\leq C_{11} \exp\Big( \langle \boldsymbol{q}_+^{(s)}, \boldsymbol{k}_+^{(s)}\rangle\Big).$$

For the last inequality, by $\eta \leq \widetilde{O}(\min\{\|\boldsymbol{\mu}\|_2^{-2}, (\sigma_p^2 d)^{-1}\} \cdot d_h^{-\frac{1}{2}})$, $\sigma_h^2 \leq \min\{\|\boldsymbol{\mu}\|_2^{-2}, (\sigma_p^2 d)^{-1}\} d_h^{-\frac{1}{2}} (\log(6N^2M^2/\delta))^{-\frac{3}{2}}$, $\langle \boldsymbol{q}_+^{(T_1)}, \boldsymbol{k}_+^{(T_1)}\rangle = o(1)$ and the monotonicity of $\langle \boldsymbol{q}_+^{(s)}, \boldsymbol{k}_+^{(s)}\rangle$ for $s \in [T_1, t]$, we have $\exp\Big( \frac{\eta C_{10}\|\boldsymbol{\mu}\|_2^4 \sigma_h^2 d_h}{\exp(\langle \boldsymbol{q}_+^{(t)}, \boldsymbol{k}_+^{(t)}\rangle)}\Big) \leq \exp(o(1)) \leq C_{11}$. Multiplying both

sides by $\left(\langle q_+^{(s+1)}, k_+^{(s+1)}\rangle - \langle q_+^{(s)}, k_+^{(s)}\rangle\right)$ simultaneously gives

$$
\begin{aligned}
&\exp(\langle q_+^{(s+1)}, k_+^{(s+1)}\rangle)\left(\langle q_+^{(s+1)}, k_+^{(s+1)}\rangle - \langle q_+^{(s)}, k_+^{(s)}\rangle\right) \\
&\leq C_{11}\exp\left(\langle q_+^{(s)}, k_+^{(s)}\rangle\right) \cdot \left(\langle q_+^{(s+1)}, k_+^{(s+1)}\rangle - \langle q_+^{(s)}, k_+^{(s)}\rangle\right) \\
&\leq \eta C_{12}\|\boldsymbol{\mu}\|_2^4\sigma_h^2 d_h,
\end{aligned}
\tag{48}
$$

where the last inequality is by plugging (46). Taking a summation we have

$$
\begin{aligned}
&\int_{\langle q_+^{(T_1)}, k_+^{(T_1)}\rangle}^{\langle q_+^{(t+1)}, k_+^{(t+1)}\rangle} \exp(x)dx \\
&\leq \sum_{s=T_1}^{t} \exp(\langle q_+^{(s+1)}, k_+^{(s+1)}\rangle)\left(\langle q_+^{(s+1)}, k_+^{(s+1)}\rangle - \langle q_+^{(s)}, k_+^{(s)}\rangle\right) \\
&\leq \sum_{s=T_1}^{t} \eta C_{12}\|\boldsymbol{\mu}\|_2^4\sigma_h^2 d_h \\
&\leq T_2 \cdot \eta C_{12}\|\boldsymbol{\mu}\|_2^4\sigma_h^2 d_h \\
&\leq \frac{d_h^{\frac{1}{2}}}{\log(6N^2M^2/\delta)^2},
\end{aligned}
\tag{49}
$$

where the first inequality is due to $\langle q_+^{(s)}, k_+^{(s)}\rangle$ is monotone increasing, the last inequality is by $T_2 = \Theta(\eta^{-1}\|\boldsymbol{\mu}\|_2^{-2}\|\boldsymbol{w}_O\|_2^{-2}\log(6N^2M^2/\delta)^{-1})$ and $\sigma_h^2 \leq \min\{\|\boldsymbol{\mu}\|_2^{-2}, (\sigma_p^2 d)^{-1}\}d_h^{-\frac{1}{2}}(\log(6N^2M^2/\delta))^{-\frac{3}{2}}$. By $\int_{\langle q_+^{(T_1)}, k_+^{(T_1)}\rangle}^{\langle q_+^{(t+1)}, k_+^{(t+1)}\rangle} \exp(x)dx = \exp(\langle q_+^{(t+1)}, k_+^{(t+1)}\rangle) - \exp(\langle q_+^{(T_1)}, k_+^{(T_1)}\rangle)$, we have

$$
\langle q_+^{(t+1)}, k_+^{(t+1)}\rangle \leq \log\left(\langle q_+^{(T_1)}, k_+^{(T_1)}\rangle + \frac{d_h^{\frac{1}{2}}}{\log(6N^2M^2/\delta)}\right) \leq \log\left(d_h^{\frac{1}{2}}\right),
\tag{50}
$$

By the results of F.6, we also have

$$
\langle q_-^{(s+1)}, k_-^{(s+1)}\rangle - \langle q_-^{(s)}, k_-^{(s)}\rangle \leq \frac{\eta C_{10}\|\boldsymbol{\mu}\|_2^4\sigma_h^2 d_h}{\exp(\langle q_-^{(s)}, k_-^{(s)}\rangle)}.
\tag{51}
$$

$$
\langle q_\pm^{(s+1)}, k_{n,j}^{(s+1)}\rangle - \langle q_\pm^{(s)}, k_{n,j}^{(s)}\rangle \geq -\frac{\eta C_{10}\sigma_p^2 d\|\boldsymbol{\mu}\|_2^2\sigma_h^2 d_h}{N} \cdot \exp(\langle q_\pm^{(s)}, k_{n,j}^{(s)}\rangle).
\tag{52}
$$

$$
\langle q_{n,i}^{(s+1)}, k_\pm^{(s+1)}\rangle - \langle q_{n,i}^{(s)}, k_\pm^{(s)}\rangle \leq \frac{\eta C_{10}\sigma_p^2 d\|\boldsymbol{\mu}\|_2^2\sigma_h^2 d_h}{N\exp(\langle q_{n,i}^{(s)}, k_\pm^{(s)}\rangle)}.
\tag{53}
$$

$$
\langle q_{n,i}^{(s+1)}, k_{n,j}^{(s+1)}\rangle - \langle q_{n,i}^{(s)}, k_{n,j}^{(s)}\rangle \geq -\frac{\eta C_{10}\sigma_p^4 d^2\sigma_h^2 d_h}{N} \cdot \exp(\langle q_{n,i}^{(s)}, k_{n,j}^{(s)}\rangle).
\tag{54}
$$

Then using the similar method as for $\langle q_+^{(t+1)}, k_+^{(t+1)}\rangle$, we get

$$
\langle q_-^{(t+1)}, k_-^{(t+1)}\rangle \leq \log\left(d_h^{\frac{1}{2}}\right),
$$

$$
\langle q_\pm^{(t+1)}, k_{n,j}^{(t+1)}\rangle \geq -\log\left(d_h^{\frac{1}{2}}\right),
$$

$$\langle \boldsymbol{q}_{n,i}^{(t+1)}, \boldsymbol{k}_{\pm}^{(t+1)} \rangle \le \log\left(d_h^{\frac{1}{2}}\right),$$

$$\langle \boldsymbol{q}_{n,i}^{(t+1)}, \boldsymbol{k}_{n,j}^{(t+1)} \rangle \ge -\log\left(d_h^{\frac{1}{2}}\right).$$

Next we provide the upper bound for $|\langle \boldsymbol{q}_{\pm}^{(t+1)}, \boldsymbol{k}_{\mp}^{(t+1)} \rangle|, |\langle \boldsymbol{q}_{n,i}^{(t+1)}, \boldsymbol{k}_{n',j}^{(t+1)} \rangle|$. By the results of F.7, we have

$$\sum_{s=T_1}^{t} |\alpha_{+,+}^{(s)}|, \sum_{s=T_1}^{t} |\alpha_{-,-}^{(s)}|, \sum_{s=T_1}^{t} |\beta_{+,+}^{(s)}|, \sum_{s=T_1}^{t} |\beta_{-,-}^{(s)}|, \sum_{s=T_1}^{t} |\beta_{n,i,+}^{(s)}|, \sum_{s=T_1}^{t} |\beta_{n,i,-}^{(s)}| = O\left(N^{\frac{1}{2}} d_h^{-\frac{1}{4}}\right),$$

(55)

for $i \in [M] \backslash \{1\}, n \in S_{\pm}$.

$$\sum_{s=T_1}^{t} |\alpha_{n,+,i}^{(s)}|, \sum_{s=T_1}^{t} |\alpha_{n,-,i}^{(s)}| = O\left(N^{-\frac{1}{2}} d_h^{-\frac{1}{4}}\right),$$

(56)

for $i \in [M] \backslash \{1\}, n \in S_{\pm}$.

$$\sum_{s=T_1}^{t} |\beta_{n,+,i}^{(s)}|, \sum_{s=T_1}^{t} |\beta_{n,-,i}^{(s)}| = O\left(\text{SNR} \cdot N^{-\frac{1}{2}} d_h^{-\frac{1}{4}}\right)$$

(57)

for $i \in [M] \backslash \{1\}, n \in S_{\pm}$.

$$\sum_{s=T_1}^{t} |\alpha_{n,i,+}^{(s)}|, \sum_{s=T_1}^{t} |\alpha_{n,i,-}^{(s)}|, \sum_{s=T_1}^{t} |\alpha_{n,i,n,j}^{(s)}|, \sum_{s=T_1}^{t} |\beta_{n,j,n,i}^{(s)}| = O\left(d_h^{-\frac{1}{4}}\right)$$

(58)

for $i, j \in [M] \backslash \{1\}, n \in S_{\pm}$.

$$\sum_{s=T_1}^{t} |\alpha_{n,i,n',j}^{(s)}|, \sum_{s=T_1}^{t} |\beta_{n,j,n',i}^{(s)}| = O\left(d^{-\frac{1}{2}} d_h^{-\frac{1}{4}} \log(6N^2 M^2/\delta)\right)$$

(59)

for $i, j \in [M]\backslash\{1\}, n, n' \in [N], n \neq n'$. Plugging these into the update rule of $\langle q_\pm^{(t)}, k_\mp^{(t)} \rangle, \langle q_{n,i}^{(t)}, k_{\bar{n},j}^{(t)} \rangle$ and assume that propositions $\mathcal{C}(T_1), \ldots, \mathcal{C}(t)$ hold, we have

$$
\begin{aligned}
|\langle q_+^{(t+1)}, k_-^{(t+1)} \rangle| &\leq |\langle q_+^{(T_1)}, k_-^{(T_1)} \rangle| + \sum_{s=T_1}^{t} |\langle q_+^{(s+1)}, k_-^{(s+1)} \rangle - \langle q_+^{(s)}, k_-^{(s)} \rangle| \\
&\leq |\langle q_+^{(T_1)}, k_-^{(T_1)} \rangle| \\
&\quad + \sum_{s=T_1}^{t} \left| \alpha_{+,+}^{(s)} \langle k_+^{(s)}, k_-^{(s)} \rangle + \sum_{n \in S_+} \sum_{i=2}^{M} \alpha_{n,+,i}^{(s)} \langle k_{n,i}^{(s)}, k_-^{(s)} \rangle \right. \\
&\quad + \beta_{-,-}^{(s)} \langle q_+^{(s)}, q_-^{(s)} \rangle + \sum_{n \in S_-} \sum_{i=2}^{M} \beta_{n,-,i}^{(s)} \langle q_{n,i}^{(s)}, q_+^{(s)} \rangle \\
&\quad + \left( \alpha_{+,+}^{(s)} k_+^{(s)} + \sum_{n \in S_+} \sum_{i=2}^{M} \alpha_{n,+,i}^{(s)} k_{n,i}^{(s)} \right) \\
&\quad \left. \cdot \left( \beta_{-,-}^{(s)} q_-^{(s)\top} + \sum_{n \in S_-} \sum_{i=2}^{M} \beta_{n,-,i}^{(s)} q_{n,i}^{(s)\top} \right) \right| \\
&\leq |\langle q_+^{(T_1)}, k_-^{(T_1)} \rangle| \\
&\quad + \sum_{s=T_1}^{t} |\alpha_{+,+}^{(s)}||\langle k_+^{(s)}, k_-^{(s)} \rangle| + \sum_{n \in S_+} \sum_{i=2}^{M} \sum_{s=T_1}^{t} |\alpha_{n,+,i}^{(s)}||\langle k_{n,i}^{(s)}, k_-^{(s)} \rangle| \\
&\quad + \sum_{s=T_1}^{t} |\beta_{-,-}^{(s)}||\langle q_+^{(s)}, q_-^{(s)} \rangle| + \sum_{n \in S_-} \sum_{i=2}^{M} \sum_{s=T_1}^{t} |\beta_{n,-,i}^{(s)}||\langle q_{n,i}^{(s)}, q_+^{(s)} \rangle| \\
&\quad + \{lower\ order\ term\} \\
&= |\langle q_+^{(T_1)}, k_-^{(T_1)} \rangle| \\
&\quad + O\left( N^{\frac{1}{2}} d_h^{-\frac{1}{4}} \right) \cdot o(1) + N \cdot M \cdot O\left( N^{-\frac{1}{2}} d_h^{-\frac{1}{4}} \right) \cdot o(1) \\
&\quad + O\left( N^{\frac{1}{2}} d_h^{-\frac{1}{4}} \right) \cdot o(1) + N \cdot M \cdot O\left( \mathrm{SNR} \cdot N^{-\frac{1}{2}} d_h^{-\frac{1}{4}} \right) \cdot o(1) \\
&= |\langle q_+^{(T_1)}, k_-^{(T_1)} \rangle| + o\left( N^{\frac{1}{2}} d_h^{-\frac{1}{4}} \right) + o\left( \mathrm{SNR} \cdot N^{\frac{1}{2}} d_h^{-\frac{1}{4}} \right) \\
&= o(1),
\end{aligned}
\tag{60}
$$

where the first inequality is by triangle inequality, the second inequality is by (213), the last equality is by $|\langle q_+^{(T_1)}, k_-^{(T_1)} \rangle| = o(1)$ and $d_h = \widetilde{\Omega}\left( \max\{\mathrm{SNR}^4, \mathrm{SNR}^{-4}\} N^2 \epsilon^{-2} \right)$. Similarly we have

$$|\langle \boldsymbol{q}_-^{(t+1)}, \boldsymbol{k}_+^{(t+1)}\rangle| = o(1).$$

$$|\langle \boldsymbol{q}_{n,i}^{(t+1)}, \boldsymbol{k}_{\overline{n},j}^{(t+1)}\rangle| \le |\langle \boldsymbol{q}_{n,i}^{(T_1)}, \boldsymbol{k}_{\overline{n},j}^{(T_1)}\rangle| + \sum_{s=T_1}^{t} |\langle \boldsymbol{q}_{n,i}^{(s+1)}, \boldsymbol{k}_{\overline{n},j}^{(s+1)}\rangle - \langle \boldsymbol{q}_{n,i}^{(s)}, \boldsymbol{k}_{\overline{n},j}^{(s)}\rangle|$$

$$\le |\langle \boldsymbol{q}_{n,i}^{(T_1)}, \boldsymbol{k}_{\overline{n},j}^{(T_1)}\rangle|$$

$$+ \sum_{s=T_1}^{t} \Big| \alpha_{n,i,+}^{(s)} \langle \boldsymbol{k}_+^{(s)}, \boldsymbol{k}_{\overline{n},j}^{(s)}\rangle + \alpha_{n,i,-}^{(s)} \langle \boldsymbol{k}_-^{(s)}, \boldsymbol{k}_{\overline{n},j}^{(s)}\rangle + \sum_{n'=1}^{N} \sum_{l=2}^{M} \alpha_{n,i,n',l}^{(s)} \langle \boldsymbol{k}_{n',l}^{(s)}, \boldsymbol{k}_{\overline{n},j}^{(s)}\rangle$$

$$+ \beta_{\overline{n},j,+}^{(s)} \langle \boldsymbol{q}_+^{(s)}, \boldsymbol{q}_{n,i}^{(s)}\rangle + \beta_{\overline{n},j,-}^{(s)} \langle \boldsymbol{q}_-^{(s)}, \boldsymbol{q}_{n,i}^{(s)}\rangle + \sum_{n'=1}^{N} \sum_{l=2}^{M} \beta_{\overline{n},j,n',l}^{(s)} \langle \boldsymbol{q}_{n',l}^{(s)}, \boldsymbol{q}_{n,i}^{(s)}\rangle$$

$$+ \left( \alpha_{n,i,+}^{(s)} \boldsymbol{k}_+^{(s)} + \alpha_{n,i,-}^{(s)} \boldsymbol{k}_-^{(s)} + \sum_{n'=1}^{N} \sum_{l=2}^{M} \alpha_{n,i,n',l}^{(s)} \boldsymbol{k}_{n',l}^{(s)} \right)$$

$$\cdot \left( \beta_{\overline{n},j,+}^{(s)} \boldsymbol{q}_+^{(s)\top} + \beta_{\overline{n},j,-}^{(s)} \boldsymbol{q}_-^{(s)\top} + \sum_{n'=1}^{N} \sum_{l=2}^{M} \beta_{\overline{n},j,n',l}^{(s)} \boldsymbol{q}_{n',l}^{(s)\top} \right) \Big|$$

$$\le |\langle \boldsymbol{q}_{n,i}^{(T_1)}, \boldsymbol{k}_{\overline{n},j}^{(T_1)}\rangle|$$

$$+ \sum_{s=T_1}^{t} |\alpha_{n,i,+}^{(s)}||\langle \boldsymbol{k}_+^{(s)}, \boldsymbol{k}_{\overline{n},j}^{(s)}\rangle| + \sum_{s=T_1}^{t} |\alpha_{n,i,-}^{(s)}||\langle \boldsymbol{k}_-^{(s)}, \boldsymbol{k}_{\overline{n},j}^{(s)}\rangle|$$

$$+ \sum_{s=T_1}^{t} |\alpha_{n,i,\overline{n},j}^{(s)}||\|\boldsymbol{k}_{\overline{n},j}^{(s)}\|_2^2 + \sum_{l=2}^{M} \sum_{s=T_1}^{t} |\alpha_{n,i,n,l}^{(s)}||\langle \boldsymbol{k}_{n,l}^{(s)}, \boldsymbol{k}_{\overline{n},j}^{(s)}\rangle|$$

$$+ \sum_{n' \ne n \wedge (l \ne j \vee n' \ne \overline{n})} \sum_{s=T_1}^{t} |\alpha_{n,i,n',l}^{(s)}||\langle \boldsymbol{k}_{n',l}^{(s)}, \boldsymbol{k}_{\overline{n},j}^{(s)}\rangle| \qquad (61)$$

$$+ \sum_{s=T_1}^{t} |\beta_{\overline{n},j,+}^{(s)}||\langle \boldsymbol{q}_+^{(s)}, \boldsymbol{q}_{n,i}^{(s)}\rangle| + \sum_{s=T_1}^{t} |\beta_{\overline{n},j,-}^{(s)}||\langle \boldsymbol{q}_-^{(s)}, \boldsymbol{q}_{n,i}^{(s)}\rangle|$$

$$+ \sum_{s=T_1}^{t} |\beta_{\overline{n},j,n,i}^{(s)}||\|\boldsymbol{q}_{n,i}^{(s)}\|_2^2 + \sum_{l=2}^{M} \sum_{s=T_1}^{t} |\beta_{\overline{n},j,\overline{n},l}^{(s)}||\langle \boldsymbol{q}_{\overline{n},l}^{(s)}, \boldsymbol{q}_{n,i}^{(s)}\rangle|$$

$$+ \sum_{n' \ne \overline{n} \wedge (l \ne i \vee n' \ne n)} \sum_{s=T_1}^{t} |\beta_{\overline{n},j,n',l}^{(s)}||\langle \boldsymbol{q}_{n',l}^{(s)}, \boldsymbol{q}_{n,i}^{(s)}\rangle|$$

$$+ \{lower\ order\ term\}$$

$$= |\langle \boldsymbol{q}_{n,i}^{(T_1)}, \boldsymbol{k}_{\overline{n},j}^{(T_1)}\rangle|$$

$$+ O\left(d_h^{-\frac{1}{4}}\right) \cdot o(1) + O\left(d^{-\frac{1}{2}} d_h^{-\frac{1}{4}} \log(6N^2 M^2/\delta)\right) \cdot \Theta(\sigma_p^2 \sigma_h^2 d d_h)$$

$$+ M \cdot O\left(d_h^{-\frac{1}{4}}\right) \cdot o(1) + N \cdot M \cdot O\left(d^{-\frac{1}{2}} d_h^{-\frac{1}{4}} \log(6N^2 M^2/\delta)\right) \cdot o(1)$$

$$+ O\left(N^{\frac{1}{2}} d_h^{-\frac{1}{4}}\right) \cdot o(1)$$

$$= |\langle \boldsymbol{q}_{n,i}^{(T_1)}, \boldsymbol{k}_{\overline{n},j}^{(T_1)}\rangle| + o\left(d_h^{-\frac{1}{4}}\right)$$

$$+ O\left(d^{-\frac{1}{2}} d_h^{\frac{1}{4}}\right) + o\left(N d^{-\frac{1}{2}} d_h^{-\frac{1}{4}} \log(6N^2 M^2/\delta)\right)$$

$$= o(1),$$

where the first inequality is by triangle inequality, the second inequality is by (196), the second equality is by $\sigma_h^2 \le \min\{\|\boldsymbol{\mu}\|_2^{-2}, (\sigma_p^2 d)^{-1}\} \cdot d_h^{-\frac{1}{2}} \cdot \left(\log(6N^2 M^2/\delta)\right)^{-\frac{3}{2}}$, the last equality is by $|\langle \boldsymbol{q}_{n,i}^{(T_1)}, \boldsymbol{k}_{\overline{n},j}^{(T_1)}\rangle| = o(1)$, $d = \widetilde{\Omega}\left(\epsilon^{-2} N^2 d_h\right)$ and $d_h = \widetilde{\Omega}\left(\max\{\text{SNR}^4, \text{SNR}^{-4}\} N^2 \epsilon^{-2}\right)$.

## D.3 Stage III

In Stage III, the outputs of ViT grow up and the loss derivatives are no longer at $o(1)$. We will carefully compute the growth rate of $V_\pm$ and $V_{n,i}$ while keeping monitoring the monotonicity of $\langle \boldsymbol{q}, \boldsymbol{k} \rangle$. By substituting $t = T_2 = \Theta\left(\frac{1}{\eta \|\boldsymbol{\mu}\|_2^2 \|\boldsymbol{w}_O\|_2^2}\right)$ into propositions $\mathcal{B}(t), \mathcal{C}(t), \mathcal{D}(t), \mathcal{E}(t)$ in Stage II, we have the following conditions at the beginning of stage III

$$|V_+^{(T_2)}|, |V_-^{(T_2)}|, |V_{n,i}^{(T_2)}| = o(1),$$

$$V_+^{(T_2)} \geq 3M \cdot |V_{n,i}^{(T_2)}|,$$

$$V_-^{(T_2)} \leq -3M \cdot |V_{n,i}^{(T_2)}|,$$

$$\|\boldsymbol{q}_\pm^{(T_2)}\|_2^2, \|\boldsymbol{k}_\pm^{(T_2)}\|_2^2 = \Theta\left(\|\boldsymbol{\mu}\|_2^2 \sigma_h^2 d_h\right),$$

$$\|\boldsymbol{q}_{n,i}^{(T_2)}\|_2^2, \|\boldsymbol{k}_{n,i}^{(T_2)}\|_2^2 = \Theta\left(\sigma_p^2 \sigma_h^2 dd_h\right),$$

$$|\langle \boldsymbol{q}_+^{(T_2)}, \boldsymbol{q}_-^{(T_2)} \rangle|, |\langle \boldsymbol{q}_\pm^{(T_2)}, \boldsymbol{q}_{n,i}^{(T_2)} \rangle|, |\langle \boldsymbol{q}_{n,i}^{(T_2)}, \boldsymbol{q}_{n',j}^{(T_2)} \rangle| = o(1),$$

$$|\langle \boldsymbol{k}_+^{(T_2)}, \boldsymbol{k}_-^{(T_2)} \rangle|, |\langle \boldsymbol{k}_\pm^{(T_2)}, \boldsymbol{k}_{n,i}^{(T_2)} \rangle|, |\langle \boldsymbol{k}_{n,i}^{(T_2)}, \boldsymbol{k}_{n',j}^{(T_2)} \rangle| = o(1),$$

for $i, j \in [M] \backslash \{1\}, n, n' \in [N], i \neq j$ or $n \neq n'$.

$$\Lambda_{n,\pm,j}^{(T_2)} \geq \log\left(\exp(\Lambda_{n,\pm,j}^{(T_1)}) + \Theta\left(\frac{d_h^{\frac{1}{2}}}{N\left(\log(6N^2M^2/\delta)\right)^3}\right)\right)$$

$$\Lambda_{n,i,\pm,j}^{(T_2)} \geq \log\left(\exp(\Lambda_{n,i,\pm,j}^{(T_1)}) + \Theta\left(\frac{\sigma_p^2 dd_h^{\frac{1}{2}}}{N\|\boldsymbol{\mu}\|_2^2\left(\log(6N^2M^2/\delta)\right)^3}\right)\right)$$

$$|\langle \boldsymbol{q}_\pm^{(T_2)}, \boldsymbol{k}_\pm^{(T_2)} \rangle|, |\langle \boldsymbol{q}_\pm^{(T_2)}, \boldsymbol{k}_{n,j}^{(T_2)} \rangle|, |\langle \boldsymbol{q}_{n,i}^{(T_2)}, \boldsymbol{k}_\pm^{(T_2)} \rangle|, |\langle \boldsymbol{q}_{n,i}^{(T_2)}, \boldsymbol{k}_{n,j}^{(T_2)} \rangle| \leq \log(d_h^{\frac{1}{2}})$$

$$|\langle \boldsymbol{q}_\pm^{(T_2)}, \boldsymbol{k}_\mp^{(T_2)} \rangle|, |\langle \boldsymbol{q}_{n,i}^{(T_2)}, \boldsymbol{k}_{\overline{n},j}^{(T_2)} \rangle| = o(1)$$

for $i, j \in [M] \backslash \{1\}, n, \overline{n} \in [N], n \neq \overline{n}$.

Let $T_3 = \Theta\left(\frac{1}{\eta \epsilon \|\boldsymbol{\mu}\|_2^2 \|\boldsymbol{w}_O\|_2^2}\right)$, Next we prove the following four propositions $\mathcal{F}(t), \mathcal{G}(t), \mathcal{H}(t), \mathcal{I}(t)$ by induction on t for $t \in [T_2, T_3]$:

- $\mathcal{F}(t)$:

$$V_+^{(t)} \geq 3M \cdot |V_{n,i}^{(t)}|,$$

$$V_-^{(t)} \leq -3M \cdot |V_{n,i}^{(t)}|,$$

$$|V_{n,i}^{(t)}| = o(1),$$

$$\log\left(\exp(V_+^{(T_2)}) + \eta C_{17} \|\boldsymbol{\mu}\|_2^2 \|\boldsymbol{w}_O\|_2^2 (t - T_2)\right) \leq V_+^{(t)} \leq 2\log\left(O(\frac{1}{\epsilon})\right),$$

$$-2\log\left(O(\frac{1}{\epsilon})\right) \leq V_-^{(t)} \leq -\log\left(\exp(-V_-^{(T_2)}) + \eta C_{17} \|\boldsymbol{\mu}\|_2^2 \|\boldsymbol{w}_O\|_2^2 (t - T_2)\right)$$

  for $i \in [M] \backslash \{1\}, n \in [N]$.

- $\mathcal{G}(t)$:

$$\|\boldsymbol{q}_\pm^{(t)}\|_2^2, \|\boldsymbol{k}_\pm^{(t)}\|_2^2 = \Theta(\|\boldsymbol{\mu}\|_2^2 \sigma_h^2 d_h),$$

$$\|\boldsymbol{q}_{n,i}^{(t)}\|_2^2, \|\boldsymbol{k}_{n,i}^{(t)}\|_2^2 = \Theta\left(\sigma_p^2 \sigma_h^2 dd_h\right),$$

$$|\langle \boldsymbol{q}_+^{(t)}, \boldsymbol{q}_-^{(t)} \rangle|, |\langle \boldsymbol{q}_\pm^{(t)}, \boldsymbol{q}_{n,i}^{(t)} \rangle|, |\langle \boldsymbol{q}_{n,i}^{(t)}, \boldsymbol{q}_{n',j}^{(t)} \rangle| = o(1),$$

$$|\langle \boldsymbol{k}_+^{(t)}, \boldsymbol{k}_-^{(t)} \rangle|, |\langle \boldsymbol{k}_\pm^{(t)}, \boldsymbol{k}_{n,i}^{(t)} \rangle|, |\langle \boldsymbol{k}_{n,i}^{(t)}, \boldsymbol{k}_{n',j}^{(t)} \rangle| = o(1)$$

  for $i, j \in [M] \backslash \{1\}, n, n' \in [N], i \neq j$ or $n \neq n'$.

- $\mathcal{H}(t)$:

$$\langle \boldsymbol{q}_\pm^{(t+1)}, \boldsymbol{k}_\pm^{(t+1)}\rangle \geq \langle \boldsymbol{q}_\pm^{(t)}, \boldsymbol{k}_\pm^{(t)}\rangle,$$

$$\langle \boldsymbol{q}_{n,i}^{(t+1)}, \boldsymbol{k}_\pm^{(t+1)}\rangle \geq \langle \boldsymbol{q}_{n,i}^{(t)}, \boldsymbol{k}_\pm^{(t)}\rangle,$$

$$\langle \boldsymbol{q}_\pm^{(t+1)}, \boldsymbol{k}_{n,j}^{(t+1)}\rangle \leq \langle \boldsymbol{q}_\pm^{(t)}, \boldsymbol{k}_{n,j}^{(t)}\rangle,$$

$$\langle \boldsymbol{q}_{n,i}^{(t+1)}, \boldsymbol{k}_{n,j}^{(t+1)}\rangle \leq \langle \boldsymbol{q}_{n,i}^{(t)}, \boldsymbol{k}_{n,j}^{(t)}\rangle$$

for $i, j \in [M] \backslash \{1\}, n \in [N]$.

- $\mathcal{I}(t)$:

$$|\langle \boldsymbol{q}_\pm^{(t)}, \boldsymbol{k}_\pm^{(t)}\rangle|, |\langle \boldsymbol{q}_\pm^{(t)}, \boldsymbol{k}_{n,j}^{(t)}\rangle|, |\langle \boldsymbol{q}_{n,i}^{(t)}, \boldsymbol{k}_\pm^{(t)}\rangle|, |\langle \boldsymbol{q}_{n,i}^{(t)}, \boldsymbol{k}_{n,j}^{(t)}\rangle| \leq \log(\epsilon^{-1} d_h^{\frac{1}{2}}),$$

$$|\langle \boldsymbol{q}_\pm^{(t)}, \boldsymbol{k}_\mp^{(t)}\rangle|, |\langle \boldsymbol{q}_{n,i}^{(t)}, \boldsymbol{k}_{\overline{n},j}^{(t)}\rangle| = o(1)$$

for $i, j \in [M] \backslash \{1\}, n, n' \in [N], n \neq \overline{n}$.

By the results of Stage II, we know that $\mathcal{F}(T_1)$, $\mathcal{G}(T_2)$, $\mathcal{I}(T_2)$ are true. To prove that $\mathcal{F}(t)$, $\mathcal{G}(t)$, $\mathcal{H}(t)$ and $\mathcal{I}(t)$ are true in stage 3, we will prove the following claims holds for $t \in [T_2, T_3]$:

**Claim 5.** $\mathcal{H}(T_2), \ldots, \mathcal{H}(t-1) \Longrightarrow \mathcal{F}(t+1)$

**Claim 6.** $\mathcal{F}(t), \mathcal{G}(t), \mathcal{H}(T_2), \ldots, \mathcal{H}(t-1) \Longrightarrow \mathcal{H}(t)$

**Claim 7.** $\mathcal{F}(T_2), \ldots, \mathcal{F}(t), \mathcal{H}(T_2), \ldots, \mathcal{H}(t-1), \mathcal{I}(T_2), \ldots, \mathcal{I}(t) \Longrightarrow \mathcal{G}(t+1)$

**Claim 8.** $\mathcal{F}(T_2), \ldots, \mathcal{F}(t), \mathcal{G}(T_2), \ldots, \mathcal{G}(t), \mathcal{H}(T_2), \ldots, \mathcal{H}(t-1) \Longrightarrow \mathcal{I}(t+1)$

### D.3.1 Proof of Claim 5

The proofs for $V_+^{(t)} \geq 3M \cdot |V_{n,i}^{(t)}|$ and $V_-^{(t)} \leq -3M \cdot |V_{n,i}^{(t)}|$ are the same as for D.2.1. Based on $\mathcal{H}(T_2), \ldots, \mathcal{H}(t)$ where $\langle \boldsymbol{q}_\pm^{(s)}, \boldsymbol{k}_\pm^{(s)}\rangle$ and $\langle \boldsymbol{q}_{n,i}^{(s)}, \boldsymbol{k}_\pm^{(s)}\rangle$ are monotonically non-decreasing and $\max_j \langle \boldsymbol{q}_\pm^{(s)}, \boldsymbol{k}_{n,j}^{(s)}\rangle, \max_j \langle \boldsymbol{q}_{n,i}^{(s)}, \boldsymbol{k}_{n,j}^{(s)}\rangle$ are monotonically non-increasing for $s \in [T_2, t-1]$, we have

$$\Lambda_{n,\pm,j}^{(s)} \geq \Lambda_{n,\pm,j}^{(T_2)} \geq \log\left(\exp(\Lambda_{n,\pm,j}^{(T_1)}) + \Theta\left(\frac{d_h^{\frac{1}{2}}}{N\left(\log(6N^2M^2/\delta)\right)^3}\right)\right), \tag{62}$$

$$\Lambda_{n,i,\pm,j}^{(s)} \geq \Lambda_{n,i,\pm,j}^{(T_2)} \geq \log\left(\exp(\Lambda_{n,i,\pm,j}^{(T_1)}) + \Theta\left(\frac{\sigma_p^2 d d_h^{\frac{1}{2}}}{N\|\boldsymbol{\mu}\|_2^2\left(\log(6N^2M^2/\delta)\right)^3}\right)\right) \tag{63}$$

for $i, j \in [M]\backslash\{1\}, n \in [N], s \in [T_2, t]$. We further get

$$\frac{\exp(\langle \boldsymbol{q}_\pm^{(s)}, \boldsymbol{k}_{n,j}^{(s)}\rangle)}{\exp(\langle \boldsymbol{q}_\pm^{(s)}, \boldsymbol{k}_\pm^{(s)}\rangle) + \sum\limits_{j'=2}^{M} \exp(\langle \boldsymbol{q}_\pm^{(s)}, \boldsymbol{k}_{n,j'}^{(s)}\rangle)}$$

$$\leq \frac{\exp(\langle \boldsymbol{q}_\pm^{(s)}, \boldsymbol{k}_{n,j}^{(s)}\rangle)}{C \exp(\langle \boldsymbol{q}_\pm^{(s)}, \boldsymbol{k}_\pm^{(s)}\rangle)}$$

$$= \frac{1}{C \exp(\Lambda_{n,\pm,j}^{(s)})}$$

$$\leq \frac{1}{C \exp(\Lambda_{n,\pm,j}^{(T_1)}) + \Theta\left(\frac{d_h^{\frac{1}{2}}}{N\left(\log(6N^2M^2/\delta)\right)^3}\right)} \tag{64}$$

$$\leq \frac{1}{\Theta\left(\frac{d_h^{\frac{1}{2}}}{N\left(\log(6N^2M^2/\delta)\right)^3}\right)}$$

$$= O\left(\frac{N\left(\log(6N^2M^2/\delta)\right)^3}{d_h^{\frac{1}{2}}}\right).$$

For the first inequality, by (62) and the monotonicity of $\langle \boldsymbol{q}^{(s)}, \boldsymbol{k}^{(s)} \rangle$ ($\langle \boldsymbol{q}_{\pm}^{(s)}, \boldsymbol{k}_{\pm}^{(s)} \rangle$ is increasing and $\langle \boldsymbol{q}_{\pm}^{(s)}, \boldsymbol{k}_{n,j}^{(s)} \rangle$ is decreasing), there exist a constant $C$ such that $C \exp(\langle \boldsymbol{q}_{\pm}^{(s)}, \boldsymbol{k}_{\pm}^{(s)} \rangle) \geq \exp(\langle \boldsymbol{q}_{\pm}^{(s)}, \boldsymbol{k}_{\pm}^{(s)} \rangle) + \sum_{j'=2}^{M} \exp(\langle \boldsymbol{q}_{\pm}^{(s)}, \boldsymbol{k}_{n,j'}^{(s)} \rangle)$. The second inequality is by plugging (62). Similarly, we have

$$
\frac{\exp(\langle \boldsymbol{q}_{n,i}^{(s)}, \boldsymbol{k}_{n,j}^{(s)} \rangle)}{\exp(\langle \boldsymbol{q}_{n,i}^{(s)}, \boldsymbol{k}_{\pm}^{(s)} \rangle) + \sum_{j'=2}^{M} \exp(\langle \boldsymbol{q}_{n,i}^{(s)}, \boldsymbol{k}_{n,j'}^{(s)} \rangle)}
$$
$$
\leq \frac{1}{C \exp(\Lambda_{n,i,\pm,j}^{(s)})} \tag{65}
$$
$$
= O\Big( \frac{N \|\boldsymbol{\mu}\|_2^2 \big(\log(6N^2 M^2/\delta)\big)^3}{\sigma_p^2 d d_h^{\frac{1}{2}}} \Big).
$$

Plugging (64) and (65) into the update rule of $\rho_{V,n,i}$ in Lemma B.2 and get

$$
|\rho_{V,n,i}^{(s+1)} - \rho_{V,n,i}^{(s)}|
$$
$$
\leq \frac{\eta}{NM} |\ell_n'^{(s)}| \cdot \|\boldsymbol{\xi}_{n,i}\|_2^2 \cdot \Big( O\Big(\frac{N\big(\log(6N^2 M^2/\delta)\big)^3}{d_h^{\frac{1}{2}}}\Big) + O\Big(\frac{N \|\boldsymbol{\mu}\|_2^2 \big(\log(6N^2 M^2/\delta)\big)^3}{\sigma_p^2 d d_h^{\frac{1}{2}}}\Big) \Big)
$$
$$
+ \frac{\eta}{NM} \sum_{n' \neq n \vee i \neq i'} |\ell_{n'}'^{(t)}| \cdot \langle \boldsymbol{\xi}_{n,i}, \boldsymbol{\xi}_{n',i'} \rangle \cdot \Big( O\Big(\frac{N\big(\log(6N^2 M^2/\delta)\big)^3}{d_h^{\frac{1}{2}}}\Big) + O\Big(\frac{N \|\boldsymbol{\mu}\|_2^2 \big(\log(6N^2 M^2/\delta)\big)^3}{\sigma_p^2 d d_h^{\frac{1}{2}}}\Big) \Big)
$$
$$
\leq \frac{3\eta \tilde{\sigma}_p^2 d}{2NM} \cdot \Big( O\Big(\frac{N\big(\log(6N^2 M^2/\delta)\big)^3}{d_h^{\frac{1}{2}}}\Big) + O\Big(\frac{N \|\boldsymbol{\mu}\|_2^2 \big(\log(6N^2 M^2/\delta)\big)^3}{\sigma_p^2 d d_h^{\frac{1}{2}}}\Big) \Big)
$$
$$
+ \frac{\eta}{NM} N \cdot M \cdot 2\tilde{\sigma}_p^2 \sqrt{d\log(4N^2 M^2/\delta)} \sqrt{d} \cdot \Big( O\Big(\frac{N\big(\log(6N^2 M^2/\delta)\big)^3}{d_h^{\frac{1}{2}}}\Big) + O\Big(\frac{N \|\boldsymbol{\mu}\|_2^2 \big(\log(6N^2 M^2/\delta)\big)^3}{\sigma_p^2 d d_h^{\frac{1}{2}}}\Big) \Big)
$$
$$
\leq \frac{2\eta}{NM} \cdot \Big( O\Big(\frac{N\sigma_p^2 d\big(\log(6N^2 M^2/\delta)\big)^3}{d_h^{\frac{1}{2}}}\Big) + O\Big(\frac{N \|\boldsymbol{\mu}\|_2^2 \big(\log(6N^2 M^2/\delta)\big)^3}{d_h^{\frac{1}{2}}}\Big) \Big)
$$
$$
= O\Big(\frac{\eta \sigma_p^2 d\big(\log(6N^2 M^2/\delta)\big)^3}{d_h^{\frac{1}{2}}} + \frac{\eta \|\boldsymbol{\mu}\|_2^2 \big(\log(6N^2 M^2/\delta)\big)^3}{d_h^{\frac{1}{2}}}\Big)
$$

(66)

where the second inequality is by Lemma C.4 and $|\ell_n'^{(t)}| \leq 1$. For the last inequality, since $d = \widetilde{\Omega}\Big(\epsilon^{-2} N^2 d_h\Big)$, we have $N \cdot M \cdot 2\tilde{\sigma}_p^2 \sqrt{d\log(4N^2 M^2/\delta)} \leq \frac{1}{2}\tilde{\sigma}_p^2 d$. By Definition B.1 and taking a summation we have

$$
|V_{n,i}^{(t+1)}| \leq |V_{n,i}^{(T_2)}| + \sum_{s=T_2}^{t} |\rho_{V,n,i}^{(s+1)} - \rho_{V,n,i}^{(s)}| \cdot \|\boldsymbol{w}_O\|_2^2
$$
$$
\leq |V_{n,i}^{(T_2)}| + T_3 \cdot |\rho_{V,n,i}^{(t+1)} - \rho_{V,n,i}^{(t)}| \cdot \|\boldsymbol{w}_O\|_2^2
$$
$$
\leq o(1) + \Theta\Big(\frac{1}{\eta\epsilon\|\boldsymbol{\mu}\|_2^2\|\boldsymbol{w}_O\|_2^2}\Big) \cdot O\Big(\frac{\eta\sigma_p^2 d\big(\log(6N^2 M^2/\delta)\big)^3}{d_h^{\frac{1}{2}}} + \frac{\eta \|\boldsymbol{\mu}\|_2^2 \big(\log(6N^2 M^2/\delta)\big)^3}{d_h^{\frac{1}{2}}}\Big) \cdot \|\boldsymbol{w}_O\|_2^2
$$
$$
= o(1) + O\Big(\frac{\sigma_p^2 d\big(\log(6N^2 M^2/\delta)\big)^3}{\epsilon\|\boldsymbol{\mu}\|_2^2 d_h^{\frac{1}{2}}} + \frac{\big(\log(6N^2 M^2/\delta)\big)^3}{\epsilon d_h^{\frac{1}{2}}}\Big)
$$
$$
= o(1) + o(1)
$$
$$
= o(1),
$$

(67)

where the first equality is by $N \cdot \mathrm{SNR}^2 \geq \Omega(1)$, the second equality is by $d_h = \widetilde{\Omega}\Big( \max\{\mathrm{SNR}^4, \mathrm{SNR}^{-4}\} N^2 \epsilon^{-2} \Big)$. Then we have a constant upper bound for the sum of $V_{n,i}$ as follows:

$$\sum_{i \in [M] \setminus \{1\}} |V_{n,i}^{(s)}| = (M-1) \cdot o(1) \leq C_{15}$$

for $n \in [N], s \in [T_2, t]$.

Expanding (64) and (65) we have

$$\frac{\exp(\langle \boldsymbol{q}_\pm^{(s)}, \boldsymbol{k}_{n,j}^{(s)} \rangle)}{\exp(\langle \boldsymbol{q}_\pm^{(s)}, \boldsymbol{k}_+^{(s)} \rangle) + \sum\limits_{j'=2}^{M} \exp(\langle \boldsymbol{q}_\pm^{(s)}, \boldsymbol{k}_{n,j'}^{(s)} \rangle)} = O\Big( \frac{N\big( \log(6N^2 M^2/\delta) \big)^3}{d_h^{\frac{1}{2}}} \Big) = o(1), \quad (68)$$

where the equality is by $d_h = \widetilde{\Omega}\Big( \max\{\mathrm{SNR}^4, \mathrm{SNR}^{-4}\} N^2 \epsilon^{-2} \Big)$.

$$\frac{\exp(\langle \boldsymbol{q}_{n,i}^{(s)}, \boldsymbol{k}_{n,j}^{(s)} \rangle)}{\exp(\langle \boldsymbol{q}_{n,i}^{(s)}, \boldsymbol{k}_+^{(s)} \rangle) + \sum\limits_{j'=2}^{M} \exp(\langle \boldsymbol{q}_{n,i}^{(s)}, \boldsymbol{k}_{n,j'}^{(s)} \rangle)} = O\Big( \frac{N\|\boldsymbol{\mu}\|_2^2\big( \log(6N^2 M^2/\delta) \big)^3}{\sigma_p^2 dd_h^{\frac{1}{2}}} \Big) = o(1), \quad (69)$$

where the equality is by $d_h = \widetilde{\Omega}\Big( \max\{\mathrm{SNR}^4, \mathrm{SNR}^{-4}\} N^2 \epsilon^{-2} \Big)$. Then we have

$$softmax(\langle \boldsymbol{q}_\pm^{(s)}, \boldsymbol{k}_\pm^{(s)} \rangle) \geq 1 - (M-1) \cdot o(1) \geq 1 - o(1), \quad (70)$$

$$softmax(\langle \boldsymbol{q}_{n,i}^{(s)}, \boldsymbol{k}_\pm^{(s)} \rangle) \geq 1 - (M-1) \cdot o(1) \geq 1 - o(1) \quad (71)$$

for $i \in [M] \setminus \{1\}, n \in [N], s \in [T_2, t]$. Next we provide the bounds for $-\ell_n'^{(s)}$. Note that $\ell(z) = \log(1 + \exp(-z))$ and $-\ell' = \exp(-z)/(1 + \exp(-z))$, without loss of generality, we assume $y_n = 1$, we have

$$-\ell'(f(\boldsymbol{X}_n, \theta(s))) = \frac{1}{1 + \exp(\frac{1}{M} \sum\limits_{l=1}^{M} \boldsymbol{\varphi}(\boldsymbol{x}_{n,l}^\top \boldsymbol{W}_Q^{(s)} \boldsymbol{W}_K^{(s)\top} (\boldsymbol{X}_n)^\top) \boldsymbol{X}_n \boldsymbol{W}_V^{(s)} \boldsymbol{w}_O)} \quad (72)$$

where $\frac{1}{M} \sum\limits_{l=1}^{M} \boldsymbol{\varphi}(\boldsymbol{x}_{n,l}^\top \boldsymbol{W}_Q^{(s)} \boldsymbol{W}_K^{(s)\top} (\boldsymbol{X}_n)^\top) \boldsymbol{X}_n \boldsymbol{W}_V^{(s)} \boldsymbol{w}_O$ can be bounded as follows:

$$\frac{1}{M} \sum_{l=1}^{M} \boldsymbol{\varphi}(\boldsymbol{x}_{n,l}^\top \boldsymbol{W}_Q^{(s)} \boldsymbol{W}_K^{(s)\top} (\boldsymbol{X}_n)^\top) \boldsymbol{X}_n \boldsymbol{W}_V^{(s)} \boldsymbol{w}_O$$

$$= \frac{1}{M} \Big( \big(softmax(\langle \boldsymbol{q}_\pm^{(s)}, \boldsymbol{k}_\pm^{(s)} \rangle)\big) + \sum_{l=2}^{M} softmax(\langle \boldsymbol{q}_{n,l}^{(s)}, \boldsymbol{k}_\pm^{(s)} \rangle) \big) \cdot \boldsymbol{\mu}_+^\top \boldsymbol{W}_V^{(s)} \boldsymbol{w}_O$$

$$+ \sum_{j \in [M] \setminus \{1\}} \big(softmax(\langle \boldsymbol{q}_\pm^{(s)}, \boldsymbol{k}_{n,j}^{(s)} \rangle) + \sum_{l=2}^{M} softmax(\langle \boldsymbol{q}_{n,l}^{(s)}, \boldsymbol{k}_{n,j}^{(s)} \rangle) \big) \cdot \boldsymbol{\xi}_{n,j}^\top \boldsymbol{W}_V^{(s)} \boldsymbol{w}_O \Big) \quad (73)$$

$$= \frac{1}{M} \Big( M \cdot \big(1 - o(1)\big) \cdot V_+^{(s)} + M \cdot o(1) \cdot \sum_{j \in [M] \setminus \{1\}} V_{n,i}^{(s)} \Big)$$

$$\geq \frac{V_+^{(s)}}{2}$$

for $s \in [T_2, t]$. The second equality is by plugging (68), (69), (70) and (71), the inequality is by $V_+^{(s)} \geq 3M \cdot |V_{n,i}^{(s)}|$ for $s \in [T_2, t]$. Similarly, we have

$$\frac{1}{M} \sum_{l=1}^{M} \boldsymbol{\varphi}(\boldsymbol{x}_{n,l}^\top \boldsymbol{W}_Q^{(s)} \boldsymbol{W}_K^{(s)\top} (\boldsymbol{X}_n)^\top) \boldsymbol{X}_n \boldsymbol{W}_V^{(s)} \boldsymbol{w}_O$$

$$\leq \max_{i \in [M] \setminus \{1\}} \{V_+^{(s)}, V_{n,i}^{(s)}\} \quad (74)$$

$$= V_+^{(s)}$$

Plugging (73) and (74) into (72) we have

$$-\ell'(f(\boldsymbol{X}_n, \theta(s))) = \frac{1}{1 + \exp(\frac{1}{M}\sum_{l=1}^{M}\varphi(\boldsymbol{x}_{n,l}^{\top}\boldsymbol{W}_Q^{(s)}\boldsymbol{W}_K^{(s)\top}(\boldsymbol{X}_n)^{\top})\boldsymbol{X}_n\boldsymbol{W}_V^{(s)}\boldsymbol{w}_O)}$$

$$\geq \frac{1}{1 + \exp(V_+^{(s)})} \tag{75}$$

$$\geq \frac{C_{16}}{\exp(V_+^{(s)})}$$

where the first inequality is by plugging (74). For the last inequality, note that $V_+^{(T_2)} \geq 0$ and $V_+^{(s)}$ is monotonically increasing, so there exist a constant $C_{16}$ such that $\frac{1}{1+\exp(V_+^{(s)})} \geq \frac{C_{16}}{\exp(V_+^{(s)})}$. We also have the upper bound

$$-\ell'(f(\boldsymbol{X}_n, \theta(s))) = \frac{1}{1 + \exp(\frac{1}{M}\sum_{l=1}^{M}\varphi(\boldsymbol{x}_{n,l}^{\top}\boldsymbol{W}_Q^{(s)}\boldsymbol{W}_K^{(s)\top}(\boldsymbol{X}_n)^{\top})\boldsymbol{X}_n\boldsymbol{W}_V^{(s)}\boldsymbol{w}_O)}$$

$$\leq \frac{1}{1 + \exp(V_+^{(s)}/2)} \tag{76}$$

$$\leq \frac{1}{\exp(V_+^{(s)}/2)}$$

Plugging (70), (71) and (75) into the update rule of $\gamma_{V,+}^{(t)}$ and in Lemma B.2 and get

$$\gamma_{V,+}^{(s+1)} - \gamma_{V,+}^{(s)}$$

$$= -\frac{\eta\|\boldsymbol{\mu}\|_2^2}{NM}\sum_{n \in S_+}\ell_n'^{(s)}\Big(\frac{\exp(\langle\boldsymbol{q}_+^{(s)}, \boldsymbol{k}_+^{(s)}\rangle)}{\exp(\langle\boldsymbol{q}_+^{(s)}, \boldsymbol{k}_+^{(s)}\rangle) + \sum_{k=2}^{M}\exp(\langle\boldsymbol{q}_+^{(s)}, \boldsymbol{k}_{n,k}^{(s)}\rangle)}$$

$$+ \sum_{j=2}^{M}\frac{\exp(\langle\boldsymbol{q}_{n,j}^{(s)}, \boldsymbol{k}_+^{(s)}\rangle)}{\exp(\langle\boldsymbol{q}_{n,j}^{(s)}, \boldsymbol{k}_+^{(s)}\rangle) + \sum_{k=2}^{M}\exp(\langle\boldsymbol{q}_{n,j}^{(s)}, \boldsymbol{k}_{n,k}^{(s)}\rangle)}\Big) \tag{77}$$

$$\geq -\frac{\eta\|\boldsymbol{\mu}\|_2^2}{NM}\sum_{n \in S_+}\ell_n'^{(s)}(M \cdot (1 - o(1)))$$

$$\geq \frac{\eta\|\boldsymbol{\mu}\|_2^2}{N} \cdot \frac{N}{4} \cdot (1 - o(1)) \cdot \frac{C_{16}}{\exp(V_+^{(s)})}$$

$$\geq \frac{\eta C_{17}\|\boldsymbol{\mu}\|_2^2}{\exp(V_+^{(s)})}$$

where the second inequality is by (75). Then by definition B.1, we get

$$V_+^{(s+1)} - V_+^{(s)} = (\gamma_{V,+}^{(s+1)} - \gamma_{V,+}^{(s)})\|\boldsymbol{w}_O\|_2^2 \geq \frac{\eta C_{17}\|\boldsymbol{\mu}\|_2^2\|\boldsymbol{w}_O\|_2^2}{\exp(V_+^{(s)})}$$

Multiply both sides simultaneously by $\exp(V_+^{(s)})$ and get

$$\exp(V_+^{(s)})(V_+^{(s+1)} - V_+^{(s)}) \geq \eta C_{17}\|\boldsymbol{\mu}\|_2^2\|\boldsymbol{w}_O\|_2^2$$

Taking a summation from $T_2$ to $t$ and get

$$
\begin{aligned}
\sum_{s=T_2}^{t} &\exp(V_+^{(s)})\big(V_+^{(s+1)} - V_+^{(s)}\big) \\
&\geq \sum_{s=T_2}^{t} \eta C_{17} \|\boldsymbol{\mu}\|_2^2 \|\boldsymbol{w}_O\|_2^2 \\
&\geq \eta C_{17} \|\boldsymbol{\mu}\|_2^2 \|\boldsymbol{w}_O\|_2^2 (t - T_2 + 1)
\end{aligned}
\tag{78}
$$

By the property that $V_+^{(s)}$ is monotonically increasing, we have

$$
\begin{aligned}
\int_{V_+^{(T_2)}}^{V_+^{(t+1)}} \exp(x)dx &\geq \sum_{s=T_2}^{t} \exp(V_+^{(s)})\big(V_+^{(s+1)} - V_+^{(s)}\big) \\
&\geq \eta C_{17} \|\boldsymbol{\mu}\|_2^2 \|\boldsymbol{w}_O\|_2^2 (t - T_2 + 1)
\end{aligned}
\tag{79}
$$

By $\int_{V_+^{(T_2)}}^{V_+^{(t+1)}} \exp(x)dx = \exp(V_+^{(t+1)}) - \exp(V_+^{(T_2)})$ we get

$$
V_+^{(t+1)} \geq \log\Big( \exp(V_+^{(T_2)}) + \eta C_{17} \|\boldsymbol{\mu}\|_2^2 \|\boldsymbol{w}_O\|_2^2 (t - T_2 + 1) \Big)
\tag{80}
$$

Similarly, we have

$$
V_-^{(t+1)} \leq -\log\Big( \exp(-V_-^{(T_2)}) + \eta C_{17} \|\boldsymbol{\mu}\|_2^2 \|\boldsymbol{w}_O\|_2^2 (t - T_2 + 1) \Big)
\tag{81}
$$

Next we provide upper bounds for $V_+^{(t+1)}$ and $V_-^{(t+1)}$. By the update rule of $\gamma_{V,+}^{(t)}$ and in Lemma B.2 we have

$$
\begin{aligned}
\gamma_{V,+}^{(s+1)} = \gamma_{V,+}^{(s)} \\
- \frac{\eta\|\boldsymbol{\mu}\|_2^2}{NM} \sum_{n\in S_+} \ell_n'^{(s)}\Big( \frac{\exp(\langle \boldsymbol{q}_+^{(s)}, \boldsymbol{k}_+^{(s)}\rangle)}{\exp(\langle \boldsymbol{q}_+^{(s)}, \boldsymbol{k}_+^{(s)}\rangle) + \sum_{k=2}^{M} \exp(\langle \boldsymbol{q}_+^{(s)}, \boldsymbol{k}_{n,k}^{(s)}\rangle)} \\
+ \sum_{j=2}^{M} \frac{\exp(\langle \boldsymbol{q}_{n,j}^{(s)}, \boldsymbol{k}_+^{(s)}\rangle)}{\exp(\langle \boldsymbol{q}_{n,j}^{(s)}, \boldsymbol{k}_+^{(s)}\rangle) + \sum_{k=2}^{M} \exp(\langle \boldsymbol{q}_{n,j}^{(s)}, \boldsymbol{k}_{n,k}^{(s)}\rangle)} \Big) \\
\leq \frac{\eta\|\boldsymbol{\mu}\|_2^2}{NM} \sum_{n\in S_+} -\ell_n'^{(s)} \cdot M \\
\leq \frac{\eta\|\boldsymbol{\mu}\|_2^2}{N} \cdot \frac{3N}{4} \cdot \frac{1}{\exp(V_+^{(s)}/2)} \\
= \frac{3\eta\|\boldsymbol{\mu}\|_2^2}{4\exp(V_+^{(s)}/2)}
\end{aligned}
$$

where the second inequality is by (76). Then by definition B.1, we get

$$
V_+^{(s+1)} - V_+^{(s)} = \big(\gamma_{V,+}^{(s+1)} - \gamma_{V,+}^{(s)}\big)\|\boldsymbol{w}_O\|_2^2 \leq \frac{3\eta\|\boldsymbol{\mu}\|_2^2\|\boldsymbol{w}_O\|_2^2}{4\exp(V_+^{(s)}/2)}
\tag{82}
$$

Further we have

$$
\begin{aligned}
\exp(V_+^{(s+1)}/2) &\leq \exp\Big(V_+^{(s)}/2 + \frac{3\eta\|\boldsymbol{\mu}\|_2^2\|\boldsymbol{w}_O\|_2^2}{8\exp(V_+^{(s)}/2)}\Big) \\
&= \exp(V_+^{(s)}/2) \cdot \exp\Big(\frac{3\eta\|\boldsymbol{\mu}\|_2^2\|\boldsymbol{w}_O\|_2^2}{8\exp(V_+^{(s)}/2)}\Big) \\
&\leq C_{18} \exp(V_+^{(s)}/2).
\end{aligned}
\tag{83}
$$

For the last inequality, by $\eta \leq \widetilde{O}(\min\{\boldsymbol{\mu}\|_2^{-2}, (\sigma_p^2 d)^{-1}\} \cdot d_h^{-\frac{1}{2}})$, $V_+^{(T_2)} = \Theta(1)$ and the monotonicity of $V_+^{(s)}$, we have $\exp(\frac{3\eta\|\boldsymbol{\mu}\|_2^2\|\boldsymbol{w}_O\|_2^2}{8\exp(V_+^{(s)}/2)}) \leq \exp(o(1)) \leq C_{18}$. Multiplying both sides by $\left(V_+^{(s+1)}/2 - V_+^{(s)}/2\right)$ simultaneously gives

$$\begin{aligned}
\exp(V_+^{(s+1)}/2)\left(V_+^{(s+1)}/2 - V_+^{(s)}/2\right) &\leq C_{18}\exp(V_+^{(s)}/2)\left(V_+^{(s+1)}/2 - V_+^{(s)}/2\right) \\
&\leq \frac{3\eta C_{18}\|\boldsymbol{\mu}\|_2^2\|\boldsymbol{w}_O\|_2^2}{8}
\end{aligned} \tag{84}$$

where the last inequality is by plugging (82). Taking a summation we have

$$\begin{aligned}
\int_{V_+^{(T_2)}/2}^{V_+^{(t+1)}/2} &\exp(x)dx \\
&\leq \sum_{s=T_2}^{t} \exp(V_+^{(s+1)}/2)\left(V_+^{(s+1)}/2 - V_+^{(s)}/2\right) \\
&\leq \sum_{s=T_2}^{T_3} \frac{3\eta C_{18}\|\boldsymbol{\mu}\|_2^2\|\boldsymbol{w}_O\|_2^2}{8} \\
&\leq \Theta\left(\frac{1}{\eta\epsilon\|\boldsymbol{\mu}\|_2^2\|\boldsymbol{w}_O\|_2^2}\right) \cdot \frac{3\eta C_{18}\|\boldsymbol{\mu}\|_2^2\|\boldsymbol{w}_O\|_2^2}{8} \\
&= O(\frac{1}{\epsilon})
\end{aligned} \tag{85}$$

By $\int_{V_+^{(T_2)}/2}^{V_+^{(t+1)}/2} \exp(x)dx = \exp(V_+^{(t+1)}/2) - \exp(V_+^{(T_2)}/2)$ we have

$$V_+^{(t+1)} \leq 2\log\left(\exp(V_+^{(T_2)}/2) + O(\frac{1}{\epsilon})\right) = 2\log\left(O(\frac{1}{\epsilon})\right)$$

Similarly, we have

$$V_-^{(t+1)} \geq -2\log\left(O(\frac{1}{\epsilon})\right)$$

### D.3.2 Proof of Claim 6

By $\mathcal{H}(T_2), \ldots, \mathcal{H}(t-1)$, we have $softmax(\langle \boldsymbol{q}_\pm^{(t)}, \boldsymbol{k}_\pm^{(t)}\rangle), softmax(\langle \boldsymbol{q}_{n,i}^{(t)}, \boldsymbol{k}_\pm^{(t)}\rangle) = 1 - o(1)$ and $softmax(\langle \boldsymbol{q}_\pm^{(t)}, \boldsymbol{k}_{n,j}^{(t)}\rangle), softmax(\langle \boldsymbol{q}_{n,i}^{(t)}, \boldsymbol{k}_{n,j}^{(t)}\rangle) = o(1)$, which have been proved in D.3.1. By the results of F.4, we have the signs of $\alpha$ and $\beta$ as follows:

$$\alpha_{+,+}^{(t)}, \alpha_{-,-}^{(t)}, \beta_{+,+}^{(t)}, \beta_{-,-}^{(t)}, \alpha_{n,i,+}^{(t)}, \alpha_{n,i,-}^{(t)}, \beta_{n,+,i}^{(t)}, \beta_{n,-,i}^{(t)} \geq 0,$$

$$\alpha_{n,+,i}^{(t)}, \alpha_{n,-,i}^{(t)}, \alpha_{n,i,n,j}^{(t)}, \beta_{n,i,+}^{(t)}, \beta_{n,i,-}^{(t)}, \beta_{n,j,n,i}^{(t)} \leq 0.$$

Then combined with $\mathcal{G}(T)$ and we have the dynamics of $\langle \boldsymbol{q}, \boldsymbol{k} \rangle$ as follows:

$$
\begin{aligned}
&\langle \boldsymbol{q}_+^{(t+1)}, \boldsymbol{k}_+^{(t+1)} \rangle - \langle \boldsymbol{q}_+^{(t)}, \boldsymbol{k}_+^{(t)} \rangle \\
&= \alpha_{+,+}^{(t)} \|\boldsymbol{k}_+^{(t)}\|_2^2 + \sum_{n \in S_+} \sum_{i=2}^{M} \alpha_{n,+,i}^{(t)} \langle \boldsymbol{k}_+^{(t)}, \boldsymbol{k}_{n,i}^{(t)} \rangle \\
&\quad + \beta_{+,+}^{(t)} \|\boldsymbol{q}_+^{(t)}\|_2^2 + \sum_{n \in S_+} \sum_{i=2}^{M} \beta_{n,+,i}^{(t)} \langle \boldsymbol{q}_+^{(t)}, \boldsymbol{q}_{n,i}^{(t)} \rangle \\
&\quad + \left( \alpha_{+,+}^{(t)} \boldsymbol{k}_+^{(t)} + \sum_{n \in S_+} \sum_{i=2}^{M} \alpha_{n,+,i}^{(t)} \boldsymbol{k}_{n,i}^{(t)} \right) \\
&\quad \cdot \left( \beta_{+,+}^{(t)} \boldsymbol{q}_+^{(t)\top} + \sum_{n \in S_+} \sum_{i=2}^{M} \beta_{n,+,i}^{(t)} \boldsymbol{q}_{n,i}^{(t)\top} \right) \\
&= \alpha_{+,+}^{(t)} \|\boldsymbol{k}_+^{(t)}\|_2^2 + \beta_{+,+}^{(t)} \|\boldsymbol{q}_+^{(t)}\|_2^2 + \{lower\ order\ term\} \\
&\geq 0
\end{aligned}
\tag{86}
$$

Similarly, we have

$$
\langle \boldsymbol{q}_-^{(t+1)}, \boldsymbol{k}_-^{(t+1)} \rangle - \langle \boldsymbol{q}_-^{(t)}, \boldsymbol{k}_-^{(t)} \rangle \geq 0.
\tag{87}
$$

$$
\begin{aligned}
&\langle \boldsymbol{q}_{n,i}^{(t+1)}, \boldsymbol{k}_+^{(t+1)} \rangle - \langle \boldsymbol{q}_{n,i}^{(t)}, \boldsymbol{k}_+^{(t)} \rangle \\
&= \alpha_{n,i,+}^{(t)} \|\boldsymbol{k}_+^{(t)}\|_2^2 + \alpha_{n,i,-}^{(t)} \langle \boldsymbol{k}_+^{(t)}, \boldsymbol{k}_-^{(t)} \rangle + \sum_{n'=1}^{N} \sum_{l=2}^{M} \alpha_{n,i,n',l}^{(t)} \langle \boldsymbol{k}_+^{(t)}, \boldsymbol{k}_{n',l}^{(t)} \rangle \\
&\quad + \beta_{+,+}^{(t)} \langle \boldsymbol{q}_+^{(t)}, \boldsymbol{q}_{n,i}^{(t)} \rangle + \sum_{n' \in S_+} \sum_{l=2}^{M} \beta_{n',+,l}^{(t)} \langle \boldsymbol{q}_{n,i}^{(t)}, \boldsymbol{q}_{n',l}^{(t)} \rangle \\
&\quad + \left( \alpha_{n,i,+}^{(t)} \boldsymbol{k}_+^{(t)} + \alpha_{n,i,-}^{(t)} \boldsymbol{k}_-^{(t)} + \sum_{n'=1}^{N} \sum_{l=2}^{M} \alpha_{n,i,n',l}^{(t)} \boldsymbol{k}_{n',l}^{(t)} \right) \\
&\quad \cdot \left( \beta_{+,+}^{(t)} \boldsymbol{q}_+^{(t)\top} + \sum_{n' \in S_+} \sum_{l=2}^{M} \beta_{n',+,l}^{(t)} \boldsymbol{q}_{n',l}^{(t)\top} \right) \\
&= \alpha_{n,i,+}^{(t)} \|\boldsymbol{k}_+^{(t)}\|_2^2 + \beta_{n,+,i}^{(t)} \|\boldsymbol{q}_{n,i}^{(t)}\|_2^2 + \{lower\ order\ term\} \\
&\geq 0.
\end{aligned}
\tag{88}
$$

Similarly, we have

$$
\langle \boldsymbol{q}_{n,i}^{(t+1)}, \boldsymbol{k}_-^{(t+1)} \rangle - \langle \boldsymbol{q}_{n,i}^{(t)}, \boldsymbol{k}_-^{(t)} \rangle \geq 0.
\tag{89}
$$

$$\langle \boldsymbol{q}_+^{(t+1)}, \boldsymbol{k}_{n,j}^{(t+1)} \rangle - \langle \boldsymbol{q}_+^{(t)}, \boldsymbol{k}_{n,j}^{(t)} \rangle$$

$$= \alpha_{+,+}^{(t)} \langle \boldsymbol{k}_+^{(t)}, \boldsymbol{k}_{n,j}^{(t)} \rangle + \sum_{n' \in S_+} \sum_{l=2}^{M} \alpha_{n',+,l}^{(t)} \langle \boldsymbol{k}_{n,j}^{(t)}, \boldsymbol{k}_{n',l}^{(t)} \rangle$$

$$+ \beta_{n,j,+}^{(t)} \|\boldsymbol{q}_+^{(t)}\|_2^2 + \beta_{n,j,-}^{(t)} \langle \boldsymbol{q}_+^{(t)}, \boldsymbol{q}_-^{(t)} \rangle + \sum_{n'=1}^{N} \sum_{l=2}^{M} \beta_{n,j,n',l}^{(t)} \langle \boldsymbol{q}_+^{(t)}, \boldsymbol{q}_{n',l}^{(t)} \rangle$$

$$+ \left( \alpha_{+,+}^{(t)} \boldsymbol{k}_+^{(t)} + \sum_{n' \in S_+} \sum_{l=2}^{M} \alpha_{n',+,l}^{(t)} \boldsymbol{k}_{n',l}^{(t)} \right) \tag{90}$$

$$\cdot \left( \beta_{n,j,+}^{(t)} \boldsymbol{q}_+^{(t)\top} + \beta_{n,j,-}^{(t)} \boldsymbol{q}_-^{(t)\top} + \sum_{n'=1}^{N} \sum_{l=2}^{M} \beta_{n,j,n',l}^{(t)} \boldsymbol{q}_{n',l}^{(t)\top} \right)$$

$$= \alpha_{n,+,j}^{(t)} \|\boldsymbol{k}_{n,j}^{(t)}\|_2^2 + \beta_{n,j,+}^{(t)} \|\boldsymbol{q}_+^{(t)}\|_2^2 + \{lower\ order\ term\}$$
$$\leq 0$$

Similarly, we have

$$\langle \boldsymbol{q}_-^{(t+1)}, \boldsymbol{k}_{n,j}^{(t+1)} \rangle - \langle \boldsymbol{q}_-^{(t)}, \boldsymbol{k}_{n,j}^{(t)} \rangle \leq 0. \tag{91}$$

$$\langle \boldsymbol{q}_{n,i}^{(t+1)}, \boldsymbol{k}_{n,j}^{(t+1)} \rangle - \langle \boldsymbol{q}_{n,i}^{(t)}, \boldsymbol{k}_{n,j}^{(t)} \rangle$$

$$= \alpha_{n,i,+}^{(t)} \langle \boldsymbol{k}_+^{(t)}, \boldsymbol{k}_{n,j}^{(t)} \rangle + \alpha_{n,i,-}^{(t)} \langle \boldsymbol{k}_-^{(t)}, \boldsymbol{k}_{n,j}^{(t)} \rangle + \sum_{n'=1}^{N} \sum_{l=2}^{M} \alpha_{n,i,n',l}^{(t)} \langle \boldsymbol{k}_{n',l}^{(t)}, \boldsymbol{k}_{n,j}^{(t)} \rangle$$

$$+ \beta_{n,j,+}^{(t)} \langle \boldsymbol{q}_+^{(t)}, \boldsymbol{q}_{n,i}^{(t)} \rangle + \beta_{n,j,-}^{(t)} \langle \boldsymbol{q}_-^{(t)}, \boldsymbol{q}_{n,i}^{(t)} \rangle + \sum_{n'=1}^{N} \sum_{l=2}^{M} \beta_{n,j,n',l}^{(t)} \langle \boldsymbol{q}_{n',l}^{(t)}, \boldsymbol{q}_{n,i}^{(t)} \rangle$$

$$+ \left( \alpha_{n,i,+}^{(t)} \boldsymbol{k}_+^{(t)} + \alpha_{n,i,-}^{(t)} \boldsymbol{k}_-^{(t)} + \sum_{n'=1}^{N} \sum_{l=2}^{M} \alpha_{n,i,n',l}^{(t)} \boldsymbol{k}_{n',l}^{(t)} \right) \tag{92}$$

$$\cdot \left( \beta_{n,j,+}^{(t)} \boldsymbol{q}_+^{(t)\top} + \beta_{n,j,-}^{(t)} \boldsymbol{q}_-^{(t)\top} + \sum_{n'=1}^{N} \sum_{l=2}^{M} \beta_{n,j,n',l}^{(t)} \boldsymbol{q}_{n',l}^{(t)\top} \right)$$

$$= \alpha_{n,i,n,j}^{(t)} \|\boldsymbol{k}_{n,j}^{(t)}\|_2^2 + \beta_{n,j,n,i}^{(t)} \|\boldsymbol{q}_{n,i}^{(t)}\|_2^2$$
$$+ \{lower\ order\ term\}$$
$$\leq 0,$$

which completes the proof. The proof for Claim 7 is in Section F.11

### D.3.3 Proof of Claim 8

By the results of F.9, we have

$$\langle \boldsymbol{q}_+^{(t+1)}, \boldsymbol{k}_+^{(t+1)} \rangle - \langle \boldsymbol{q}_+^{(t)}, \boldsymbol{k}_+^{(t)} \rangle \leq \frac{\eta C_{10} \|\boldsymbol{\mu}\|_2^4 \sigma_h^2 d_h \log\left(O(\frac{1}{\epsilon})\right)}{\exp(\langle \boldsymbol{q}_+^{(t)}, \boldsymbol{k}_+^{(t)} \rangle)}, \tag{93}$$

Further we have

$$\exp(\langle \boldsymbol{q}_+^{(t+1)}, \boldsymbol{k}_+^{(t+1)} \rangle) \leq \exp\left( \langle \boldsymbol{q}_+^{(t)}, \boldsymbol{k}_+^{(t)} \rangle + \frac{\eta C_{10} \|\boldsymbol{\mu}\|_2^4 \sigma_h^2 d_h \log\left(O(\frac{1}{\epsilon})\right)}{\exp(\langle \boldsymbol{q}_+^{(t)}, \boldsymbol{k}_+^{(t)} \rangle)} \right)$$

$$= \exp\left( \langle \boldsymbol{q}_+^{(t)}, \boldsymbol{k}_+^{(t)} \rangle \right) \cdot \exp\left( \frac{\eta C_{10} \|\boldsymbol{\mu}\|_2^4 \sigma_h^2 d_h \log\left(O(\frac{1}{\epsilon})\right)}{\exp(\langle \boldsymbol{q}_+^{(t)}, \boldsymbol{k}_+^{(t)} \rangle)} \right) \tag{94}$$

$$\leq C_{11} \exp\left( \langle \boldsymbol{q}_+^{(t)}, \boldsymbol{k}_+^{(t)} \rangle \right).$$

For the last inequality, by $\eta \leq \widetilde{O}(\min\{\|\boldsymbol{\mu}\|_2^{-2}, (\sigma_p^2 d)^{-1}\} \cdot d_h^{-\frac{1}{2}})$, $\sigma_h^2 \leq \min\{\|\boldsymbol{\mu}\|_2^{-2}, (\sigma_p^2 d)^{-1}\} \cdot d_h^{-\frac{1}{2}} \cdot \left(\log(6N^2 M^2/\delta)\right)^{-\frac{3}{2}}$, $\langle \boldsymbol{q}_+^{(T_1)}, \boldsymbol{k}_+^{(T_1)} \rangle = o(1)$ and the monotonicity of $\langle \boldsymbol{q}_+^{(s)}, \boldsymbol{k}_+^{(s)} \rangle$ for $s \in [T_1, t]$, we have $\exp\left(\frac{\eta C_{10}\|\boldsymbol{\mu}\|_2^2 \sigma_h^2 d_h \log\left(O(\frac{1}{\epsilon})\right)}{\exp(\langle \boldsymbol{q}_+^{(t)}, \boldsymbol{k}_+^{(t)} \rangle)}\right) \leq \exp(o(1)) \leq C_{11}$. Multiplying both sides by $\left(\langle \boldsymbol{q}_+^{(t+1)}, \boldsymbol{k}_+^{(t+1)} \rangle - \langle \boldsymbol{q}_+^{(t)}, \boldsymbol{k}_+^{(t)} \rangle\right)$ simultaneously gives

$$
\begin{aligned}
&\exp(\langle \boldsymbol{q}_+^{(t+1)}, \boldsymbol{k}_+^{(t+1)} \rangle)\left(\langle \boldsymbol{q}_+^{(t+1)}, \boldsymbol{k}_+^{(t+1)} \rangle - \langle \boldsymbol{q}_+^{(t)}, \boldsymbol{k}_+^{(t)} \rangle\right) \\
&\leq C_{11} \exp\left(\langle \boldsymbol{q}_+^{(t)}, \boldsymbol{k}_+^{(t)} \rangle\right) \cdot \left(\langle \boldsymbol{q}_+^{(t+1)}, \boldsymbol{k}_+^{(t+1)} \rangle - \langle \boldsymbol{q}_+^{(t)}, \boldsymbol{k}_+^{(t)} \rangle\right) \\
&\leq \eta C_{12} \|\boldsymbol{\mu}\|_2^4 \sigma_h^2 d_h \log\left(O(\frac{1}{\epsilon})\right),
\end{aligned}
\tag{95}
$$

where the last inequality is by plugging (93). Taking a summation we have

$$
\begin{aligned}
&\int_{\langle \boldsymbol{q}_+^{(T_2)}, \boldsymbol{k}_+^{(T_2)} \rangle}^{\langle \boldsymbol{q}_+^{(t+1)}, \boldsymbol{k}_+^{(t+1)} \rangle} \exp(x)dx \\
&\leq \sum_{s=T_2}^{t} \exp(\langle \boldsymbol{q}_+^{(s+1)}, \boldsymbol{k}_+^{(s+1)} \rangle)\left(\langle \boldsymbol{q}_+^{(s+1)}, \boldsymbol{k}_+^{(s+1)} \rangle - \langle \boldsymbol{q}_+^{(s)}, \boldsymbol{k}_+^{(s)} \rangle\right) \\
&\leq \sum_{s=T_2}^{t} \eta C_{12} \|\boldsymbol{\mu}\|_2^4 \sigma_h^2 d_h \log\left(O(\frac{1}{\epsilon})\right) \\
&\leq T_3 \cdot \eta C_{12} \|\boldsymbol{\mu}\|_2^4 \sigma_h^2 d_h \log\left(O(\frac{1}{\epsilon})\right) \\
&= O\left(\frac{d_h^{\frac{1}{2}} \log\left(O(\frac{1}{\epsilon})\right)}{\epsilon(\log(6N^2 M^2/\delta))^{\frac{3}{2}}}\right)
\end{aligned}
\tag{96}
$$

where the first inequality is due to $\langle \boldsymbol{q}_+^{(s)}, \boldsymbol{k}_+^{(s)} \rangle$ is monotone increasing, the last equality is by $T_3 = \Theta(\eta^{-1}\epsilon^{-1}\|\boldsymbol{\mu}\|_2^{-2}\|\boldsymbol{w}_O\|_2^{-2})$, $\|\boldsymbol{w}_O\|_2^2 = \Theta(1)$ and $\sigma_h^2 \leq \min\{\|\boldsymbol{\mu}\|_2^{-2}, (\sigma_p^2 d)^{-1}\} \cdot d_h^{-\frac{1}{2}} \cdot \left(\log(6N^2 M^2/\delta)\right)^{-\frac{3}{2}}$. By $\int_{\langle \boldsymbol{q}_+^{(T_2)}, \boldsymbol{k}_+^{(T_2)} \rangle}^{\langle \boldsymbol{q}_+^{(t+1)}, \boldsymbol{k}_+^{(t+1)} \rangle} \exp(x)dx = \exp(\langle \boldsymbol{q}_+^{(t+1)}, \boldsymbol{k}_+^{(t+1)} \rangle) - \exp(\langle \boldsymbol{q}_+^{(T_2)}, \boldsymbol{k}_+^{(T_2)} \rangle)$, we have

$$
\langle \boldsymbol{q}_+^{(t+1)}, \boldsymbol{k}_+^{(t+1)} \rangle \leq \log\left(\exp(\langle \boldsymbol{q}_+^{(T_2)}, \boldsymbol{k}_+^{(T_2)} \rangle) + O\left(\frac{d_h^{\frac{1}{2}} \log\left(O(\frac{1}{\epsilon})\right)}{\epsilon(\log(6N^2 M^2/\delta))^{\frac{3}{2}}}\right)\right) \leq \log(\epsilon^{-1} d_h^{\frac{1}{2}}),
\tag{97}
$$

where the last inequality is by $\langle \boldsymbol{q}_+^{(T_2)}, \boldsymbol{k}_+^{(T_2)} \rangle \leq \log\left(d_h^{\frac{1}{2}}\right)$. By the results of F.9, we also have

$$
\langle \boldsymbol{q}_-^{(t+1)}, \boldsymbol{k}_-^{(t+1)} \rangle - \langle \boldsymbol{q}_-^{(t)}, \boldsymbol{k}_-^{(t)} \rangle \leq \frac{\eta C_{10}\|\boldsymbol{\mu}\|_2^4 \sigma_h^2 d_h \log\left(O(\frac{1}{\epsilon})\right)}{\exp(\langle \boldsymbol{q}_-^{(t)}, \boldsymbol{k}_-^{(t)} \rangle)}.
\tag{98}
$$

$$
\langle \boldsymbol{q}_\pm^{(t+1)}, \boldsymbol{k}_{n,j}^{(t+1)} \rangle - \langle \boldsymbol{q}_\pm^{(t)}, \boldsymbol{k}_{n,j}^{(t)} \rangle \geq -\frac{\eta C_{10}\sigma_p^2 d\|\boldsymbol{\mu}\|_2^2 \sigma_h^2 d_h \log\left(O(\frac{1}{\epsilon})\right)}{N} \cdot \exp(\langle \boldsymbol{q}_\pm^{(t)}, \boldsymbol{k}_{n,j}^{(t)} \rangle).
\tag{99}
$$

$$
\langle \boldsymbol{q}_{n,i}^{(t+1)}, \boldsymbol{k}_\pm^{(t+1)} \rangle - \langle \boldsymbol{q}_{n,i}^{(t)}, \boldsymbol{k}_\pm^{(t)} \rangle \leq \frac{\eta C_{10}\sigma_p^2 d\|\boldsymbol{\mu}\|_2^2 \sigma_h^2 d_h \log\left(O(\frac{1}{\epsilon})\right)}{N \exp(\langle \boldsymbol{q}_{n,i}^{(t)}, \boldsymbol{k}_\pm^{(t)} \rangle)}.
\tag{100}
$$

$$
\langle \boldsymbol{q}_{n,i}^{(t+1)}, \boldsymbol{k}_{n,j}^{(t+1)} \rangle - \langle \boldsymbol{q}_{n,i}^{(t)}, \boldsymbol{k}_{n,j}^{(t)} \rangle \geq -\frac{\eta C_{10}\sigma_p^4 d^2 \sigma_h^2 d_h \log\left(O(\frac{1}{\epsilon})\right)}{N} \cdot \exp(\langle \boldsymbol{q}_{n,i}^{(t)}, \boldsymbol{k}_{n,j}^{(t)} \rangle).
\tag{101}
$$

Then using the similar method as for $\langle \boldsymbol{q}_+^{(t+1)}, \boldsymbol{k}_+^{(t+1)} \rangle$, we get

$$\langle \boldsymbol{q}_-^{(t+1)}, \boldsymbol{k}_-^{(t+1)} \rangle \leq \log(\epsilon^{-1} d_h^{\frac{1}{2}}),$$

$$\langle \boldsymbol{q}_\pm^{(t+1)}, \boldsymbol{k}_{n,j}^{(t+1)} \rangle \geq -\log(\epsilon^{-1} d_h^{\frac{1}{2}}),$$

$$\langle \boldsymbol{q}_{n,i}^{(t+1)}, \boldsymbol{k}_\pm^{(t+1)} \rangle \leq \log(\epsilon^{-1} d_h^{\frac{1}{2}}),$$

$$\langle \boldsymbol{q}_{n,i}^{(t+1)}, \boldsymbol{k}_{n,j}^{(t+1)} \rangle \geq -\log(\epsilon^{-1} d_h^{\frac{1}{2}}),$$

Next we provide the upper bound for $|\langle \boldsymbol{q}_\pm^{(t+1)}, \boldsymbol{k}_\mp^{(t+1)} \rangle|, |\langle \boldsymbol{q}_{n,i}^{(t+1)}, \boldsymbol{k}_{n',j}^{(t+1)} \rangle|$. By the results of F.10, we have

$$\sum_{s=T_2}^{t} |\beta_{n,+,i}^{(t)}|, \sum_{s=T_2}^{t} |\beta_{n,-,i}^{(t)}| = O\Big(\frac{\mathrm{SNR}^2 \big(\log(6N^2 M^2/\delta)\big)^3 \log\big(O(\frac{1}{\epsilon})\big)}{\epsilon d_h^{\frac{1}{2}}}\Big), \tag{102}$$

for $i \in [M]\backslash\{1\}, n \in S_\pm$.

$$\sum_{s=T_2}^{t} |\alpha_{+,+}^{(t)}|, \sum_{s=T_2}^{t} |\alpha_{-,-}^{(t)}|, \sum_{s=T_2}^{t} |\beta_{+,+}^{(t)}|, \sum_{s=T_2}^{t} |\beta_{-,-}^{(t)}|, \sum_{s=T_2}^{t} |\beta_{n,i,+}^{(t)}|, \sum_{s=T_2}^{t} |\beta_{n,i,-}^{(t)}|$$
$$= O\Big(\frac{N \big(\log(6N^2 M^2/\delta)\big)^3 \log\big(O(\frac{1}{\epsilon})\big)}{\epsilon d_h^{\frac{1}{2}}}\Big), \tag{103}$$

for $i \in [M]\backslash\{1\}, n \in S_\pm$.

$$\sum_{s=T_2}^{t} |\alpha_{n,+,i}^{(t)}|, \sum_{s=T_2}^{t} |\alpha_{n,-,i}^{(t)}|, \sum_{s=T_2}^{t} |\alpha_{n,i,+}^{(t)}|, \sum_{s=T_2}^{t} |\alpha_{n,i,-}^{(t)}|, \sum_{s=T_2}^{t} |\alpha_{n,i,n,j}^{(t)}|, \sum_{s=T_2}^{t} |\beta_{n,j,n,i}^{(t)}|$$
$$= O\Big(\frac{\big(\log(6N^2 M^2/\delta)\big)^3 \log\big(O(\frac{1}{\epsilon})\big)}{\epsilon d_h^{\frac{1}{2}}}\Big), \tag{104}$$

for $i, j \in [M]\backslash\{1\}, n \in S_\pm$.

$$\sum_{s=T_2}^{t} |\alpha_{n,i,n',j}^{(t)}|, \sum_{s=T_2}^{t} |\beta_{n,j,n',i}^{(t)}| = O\Big(\frac{\big(\log(6N^2 M^2/\delta)\big)^4 \log\big(O(\frac{1}{\epsilon})\big)}{\epsilon d^{\frac{1}{2}} d_h^{\frac{1}{2}}}\Big) \tag{105}$$

for $i, j \in [M] \backslash \{1\}, n, n' \in [N], n \neq n'$. Plugging these and proposition $\mathcal{G}(t)$ into the update rule of $|\langle \boldsymbol{q}_\pm^{(t)}, \boldsymbol{k}_\mp^{(t)} \rangle|, |\langle \boldsymbol{q}_{n,i}^{(t)}, \boldsymbol{k}_{\bar{n},j}^{(t)} \rangle|$ and get

$$|\langle \boldsymbol{q}_+^{(t+1)}, \boldsymbol{k}_-^{(t+1)} \rangle| \leq |\langle \boldsymbol{q}_+^{(T_2)}, \boldsymbol{k}_-^{(T_2)} \rangle| + \sum_{s=T_2}^{t} |\langle \boldsymbol{q}_+^{(s+1)}, \boldsymbol{k}_-^{(s+1)} \rangle - \langle \boldsymbol{q}_+^{(s)}, \boldsymbol{k}_-^{(s)} \rangle|$$

$$\leq |\langle \boldsymbol{q}_+^{(T_2)}, \boldsymbol{k}_-^{(T_2)} \rangle|$$

$$+ \sum_{s=T_2}^{t} \Big| \alpha_{+,+}^{(s)} \langle \boldsymbol{k}_+^{(s)}, \boldsymbol{k}_-^{(s)} \rangle + \sum_{n \in S_+} \sum_{i=2}^{M} \alpha_{n,+,i}^{(s)} \langle \boldsymbol{k}_{n,i}^{(s)}, \boldsymbol{k}_-^{(s)} \rangle$$

$$+ \beta_{-,-}^{(s)} \langle \boldsymbol{q}_+^{(s)}, \boldsymbol{q}_-^{(s)} \rangle + \sum_{n \in S_-} \sum_{i=2}^{M} \beta_{n,-,i}^{(s)} \langle \boldsymbol{q}_{n,i}^{(s)}, \boldsymbol{q}_+^{(s)} \rangle$$

$$+ \Big( \alpha_{+,+}^{(s)} \boldsymbol{k}_+^{(s)} + \sum_{n \in S_+} \sum_{i=2}^{M} \alpha_{n,+,i}^{(s)} \boldsymbol{k}_{n,i}^{(s)} \Big)$$

$$\cdot \Big( \beta_{-,-}^{(s)} \boldsymbol{q}_-^{(s)\top} + \sum_{n \in S_-} \sum_{i=2}^{M} \beta_{n,-,i}^{(s)} \boldsymbol{q}_{n,i}^{(s)\top} \Big) \Big|$$

$$\leq |\langle \boldsymbol{q}_+^{(T_2)}, \boldsymbol{k}_-^{(T_2)} \rangle|$$

$$+ \sum_{s=T_2}^{t} |\alpha_{+,+}^{(t)}| |\langle \boldsymbol{k}_+^{(t)}, \boldsymbol{k}_-^{(t)} \rangle| + \sum_{n \in S_+} \sum_{i=2}^{M} \sum_{s=T_2}^{t} |\alpha_{n,+,i}^{(t)}| |\langle \boldsymbol{k}_{n,i}^{(t)}, \boldsymbol{k}_-^{(t)} \rangle|$$

$$+ \sum_{s=T_2}^{t} |\beta_{-,-}^{(t)}| |\langle \boldsymbol{q}_+^{(t)}, \boldsymbol{q}_-^{(t)} \rangle| + \sum_{n \in S_-} \sum_{i=2}^{M} \sum_{s=T_2}^{t} |\beta_{n,-,i}^{(t)}| |\langle \boldsymbol{q}_{n,i}^{(t)}, \boldsymbol{q}_+^{(t)} \rangle|$$

$$+ \{lower\ order\ term\}$$

$$= |\langle \boldsymbol{q}_+^{(T_2)}, \boldsymbol{k}_-^{(T_2)} \rangle|$$

$$+ O\Big( \frac{N \big( \log(6N^2 M^2/\delta) \big)^3 \log \big( O(\tfrac{1}{\epsilon}) \big)}{\epsilon d_h^{\frac{1}{2}}} \Big) \cdot o(1) + N \cdot M \cdot O\Big( \frac{\big( \log(6N^2 M^2/\delta) \big)^3 \log \big( O(\tfrac{1}{\epsilon}) \big)}{\epsilon d_h^{\frac{1}{2}}} \Big) \cdot o(1)$$

$$+ O\Big( \frac{N \big( \log(6N^2 M^2/\delta) \big)^3 \log \big( O(\tfrac{1}{\epsilon}) \big)}{\epsilon d_h^{\frac{1}{2}}} \Big) \cdot o(1) + N \cdot M \cdot O\Big( \frac{\mathrm{SNR}^2 \big( \log(6N^2 M^2/\delta) \big)^3 \log \big( O(\tfrac{1}{\epsilon}) \big)}{\epsilon d_h^{\frac{1}{2}}} \Big) \cdot o(1)$$

$$= |\langle \boldsymbol{q}_+^{(T_2)}, \boldsymbol{k}_-^{(T_2)} \rangle| + o\Big( \frac{N \big( \log(6N^2 M^2/\delta) \big)^3 \log \big( O(\tfrac{1}{\epsilon}) \big)}{\epsilon d_h^{\frac{1}{2}}} \Big) + o\Big( \frac{N \cdot \mathrm{SNR}^2 \big( \log(6N^2 M^2/\delta) \big)^3 \log \big( O(\tfrac{1}{\epsilon}) \big)}{\epsilon d_h^{\frac{1}{2}}} \Big)$$

$$= o(1),$$

$$\tag{106}$$

where the first inequality is by triangle inequality, the second inequality is by (213), the last equality is by $|\langle \boldsymbol{q}_+^{(T_2)}, \boldsymbol{k}_-^{(T_2)} \rangle| = o(1)$ and $d_h = \widetilde{\Omega}\Big( \max\{\mathrm{SNR}^4, \mathrm{SNR}^{-4}\} N^2 \epsilon^{-2} \Big)$. Similarly we have

$$|\langle \boldsymbol{q}_-^{(t+1)}, \boldsymbol{k}_+^{(t+1)}\rangle| = o(1).$$

$$|\langle \boldsymbol{q}_{n,i}^{(t+1)}, \boldsymbol{k}_{\overline{n},j}^{(t+1)}\rangle| \leq |\langle \boldsymbol{q}_{n,i}^{(T_2)}, \boldsymbol{k}_{\overline{n},j}^{(T_2)}\rangle| + \sum_{s=T_2}^{t} |\langle \boldsymbol{q}_{n,i}^{(s+1)}, \boldsymbol{k}_{\overline{n},j}^{(s+1)}\rangle - \langle \boldsymbol{q}_{n,i}^{(s)}, \boldsymbol{k}_{\overline{n},j}^{(s)}\rangle|$$

$$\leq |\langle \boldsymbol{q}_{n,i}^{(T_2)}, \boldsymbol{k}_{\overline{n},j}^{(T_2)}\rangle|$$

$$+ \sum_{s=T_2}^{t} \Big| \alpha_{n,i,+}^{(s)}\langle \boldsymbol{k}_+^{(s)}, \boldsymbol{k}_{\overline{n},j}^{(s)}\rangle + \alpha_{n,i,-}^{(s)}\langle \boldsymbol{k}_-^{(s)}, \boldsymbol{k}_{\overline{n},j}^{(s)}\rangle + \sum_{n'=1}^{N}\sum_{l=2}^{M} \alpha_{n,i,n',l}^{(s)}\langle \boldsymbol{k}_{n',l}^{(s)}, \boldsymbol{k}_{\overline{n},j}^{(s)}\rangle$$

$$+ \beta_{\overline{n},j,+}^{(s)}\langle \boldsymbol{q}_+^{(s)}, \boldsymbol{q}_{n,i}^{(s)}\rangle + \beta_{\overline{n},j,-}^{(s)}\langle \boldsymbol{q}_-^{(s)}, \boldsymbol{q}_{n,i}^{(s)}\rangle + \sum_{n'=1}^{N}\sum_{l=2}^{M} \beta_{\overline{n},j,n',l}^{(s)}\langle \boldsymbol{q}_{n',l}^{(s)}, \boldsymbol{q}_{n,i}^{(s)}\rangle$$

$$+ \Big( \alpha_{n,i,+}^{(s)}\boldsymbol{k}_+^{(s)} + \alpha_{n,i,-}^{(s)}\boldsymbol{k}_-^{(s)} + \sum_{n'=1}^{N}\sum_{l=2}^{M} \alpha_{n,i,n',l}^{(s)}\boldsymbol{k}_{n',l}^{(s)} \Big)$$

$$\cdot \Big( \beta_{\overline{n},j,+}^{(s)}\boldsymbol{q}_+^{(s)\top} + \beta_{\overline{n},j,-}^{(s)}\boldsymbol{q}_-^{(s)\top} + \sum_{n'=1}^{N}\sum_{l=2}^{M} \beta_{\overline{n},j,n',l}^{(s)}\boldsymbol{q}_{n',l}^{(s)\top} \Big) \Big|$$

$$\leq |\langle \boldsymbol{q}_{n,i}^{(T_2)}, \boldsymbol{k}_{\overline{n},j}^{(T_2)}\rangle|$$

$$+ \sum_{s=T_2}^{t} |\alpha_{n,i,+}^{(t)}||\langle \boldsymbol{k}_+^{(t)}, \boldsymbol{k}_{\overline{n},j}^{(t)}\rangle| + \sum_{s=T_2}^{t} |\alpha_{n,i,-}^{(t)}||\langle \boldsymbol{k}_-^{(t)}, \boldsymbol{k}_{\overline{n},j}^{(t)}\rangle|$$

$$+ \sum_{s=T_2}^{t} |\alpha_{n,i,\overline{n},j}^{(t)}|\|\boldsymbol{k}_{\overline{n},j}^{(t)}\|_2^2 + \sum_{l=2}^{M}\sum_{s=T_2}^{t} |\alpha_{n,i,n,l}^{(t)}||\langle \boldsymbol{k}_{n,l}^{(t)}, \boldsymbol{k}_{\overline{n},j}^{(t)}\rangle|$$

$$+ \sum_{n'\neq n \wedge (l\neq j \vee n'\neq \overline{n})} \sum_{t=T_2}^{T_3} |\alpha_{n,i,n',l}^{(t)}||\langle \boldsymbol{k}_{n',l}^{(t)}, \boldsymbol{k}_{\overline{n},j}^{(t)}\rangle|$$

$$+ \sum_{s=T_2}^{t} |\beta_{\overline{n},j,+}^{(t)}||\langle \boldsymbol{q}_+^{(t)}, \boldsymbol{q}_{n,i}^{(t)}\rangle| + \sum_{s=T_2}^{t} |\beta_{\overline{n},j,-}^{(t)}||\langle \boldsymbol{q}_-^{(t)}, \boldsymbol{q}_{n,i}^{(t)}\rangle|$$

$$+ \sum_{s=T_2}^{t} |\beta_{\overline{n},j,n,i}^{(t)}|\|\boldsymbol{q}_{n,i}^{(t)}\|_2^2 + \sum_{l=2}^{M}\sum_{s=T_2}^{t} |\beta_{\overline{n},j,\overline{n},l}^{(t)}||\langle \boldsymbol{q}_{\overline{n},l}^{(t)}, \boldsymbol{q}_{n,i}^{(t)}\rangle|$$

$$+ \sum_{n'\neq \overline{n} \wedge (l\neq i \vee n'\neq n)} \sum_{s=T_2}^{t} |\beta_{\overline{n},j,n',l}^{(t)}||\langle \boldsymbol{q}_{n',l}^{(t)}, \boldsymbol{q}_{n,i}^{(t)}\rangle|$$

$$+ \{lower\ order\ term\}$$

$$= |\langle \boldsymbol{q}_{n,i}^{(T_2)}, \boldsymbol{k}_{\overline{n},j}^{(T_2)}\rangle|$$

$$+ O\Big( \frac{\big(\log(6N^2M^2/\delta)\big)^3 \log\big(O(\tfrac{1}{\epsilon})\big)}{\epsilon d_h^{\frac{1}{2}}} \Big) \cdot o(1) + O\Big( \frac{\big(\log(6N^2M^2/\delta)\big)^4 \log\big(O(\tfrac{1}{\epsilon})\big)}{\epsilon d^{\frac{1}{2}} d_h^{\frac{1}{2}}} \Big) \cdot \Theta(\sigma_p^2 \sigma_h^2 d d_h)$$

$$+ M \cdot O\Big( \frac{\big(\log(6N^2M^2/\delta)\big)^3 \log\big(O(\tfrac{1}{\epsilon})\big)}{\epsilon d_h^{\frac{1}{2}}} \Big) \cdot o(1) + N \cdot M \cdot O\Big( \frac{\big(\log(6N^2M^2/\delta)\big)^4 \log\big(O(\tfrac{1}{\epsilon})\big)}{\epsilon d^{\frac{1}{2}} d_h^{\frac{1}{2}}} \Big) \cdot o(1)$$

$$+ O\Big( \frac{N\big(\log(6N^2M^2/\delta)\big)^3 \log\big(O(\tfrac{1}{\epsilon})\big)}{\epsilon d_h^{\frac{1}{2}}} \Big) \cdot o(1)$$

$$= |\langle \boldsymbol{q}_{n,i}^{(T_2)}, \boldsymbol{k}_{\overline{n},j}^{(T_2)}\rangle| + o\Big( N\frac{\big(\log(6N^2M^2/\delta)\big)^3 \log\big(O(\tfrac{1}{\epsilon})\big)}{\epsilon d_h^{\frac{1}{2}}} \Big)$$

$$+ O\Big( \frac{\big(\log(6N^2M^2/\delta)\big)^3 \log\big(O(\tfrac{1}{\epsilon})\big)}{\epsilon d^{\frac{1}{2}}} \Big) + o\Big( \frac{N\big(\log(6N^2M^2/\delta)\big)^4 \log\big(O(\tfrac{1}{\epsilon})\big)}{\epsilon d^{\frac{1}{2}} d_h^{\frac{1}{2}}} \Big)$$

$$= o(1),$$

56

$$(107)$$

where the first inequality is by triangle inequality, the second inequality is by (196), the second equality is by $\sigma_h^2 \leq \min\{\|\boldsymbol{\mu}\|_2^{-2}, (\sigma_p^2 d)^{-1}\} \cdot d_h^{-\frac{1}{2}} \cdot \big(\log(6N^2M^2/\delta)\big)^{-\frac{3}{2}}$, the last equality is by $|\langle \boldsymbol{q}_{n,i}^{(T_2)}, \boldsymbol{k}_{\bar{n},j}^{(T_2)}\rangle| = o(1)$, $d = \widetilde{\Omega}\big(\epsilon^{-2} N^2 d_h\big)$ and $d_h = \widetilde{\Omega}\big(\max\{\mathrm{SNR}^4, \mathrm{SNR}^{-4}\} N^2 \epsilon^{-2}\big)$.

**Lemma D.7** (Convergence of Training Loss). *There exist $T = \frac{C_{19}}{\eta \epsilon \|\boldsymbol{\mu}\|_2^2 \|\boldsymbol{w}_O\|_2^2}$ such that*

$$L_S(\theta(T)) \leq \epsilon$$

*Proof of Lemma D.7.* Recall (68), (69), (70) and (68), we have

$$softmax(\langle \boldsymbol{q}_{\pm}^{(T)}, \boldsymbol{k}_{n,j}^{(t)}\rangle) = O\Big(\frac{N\big(\log(6N^2M^2/\delta)\big)^3}{d_h^{\frac{1}{2}}}\Big), \tag{108}$$

$$softmax(\langle \boldsymbol{q}_{n,i}^{(t)}, \boldsymbol{k}_{n,j}^{(t)}\rangle) = O\Big(\frac{N\|\boldsymbol{\mu}\|_2^2\big(\log(6N^2M^2/\delta)\big)^3}{\sigma_p^2 d d_h^{\frac{1}{2}}}\Big) = O\Big(\frac{N \cdot \mathrm{SNR}^2 \cdot \big(\log(6N^2M^2/\delta)\big)^3}{d_h^{\frac{1}{2}}}\Big), \tag{109}$$

$$softmax(\langle \boldsymbol{q}_{\pm}^{(t)}, \boldsymbol{k}_{\pm}^{(t)}\rangle) = 1 - O\Big(\frac{N\big(\log(6N^2M^2/\delta)\big)^3}{d_h^{\frac{1}{2}}}\Big), \tag{110}$$

$$softmax(\langle \boldsymbol{q}_{n,i}^{(t)}, \boldsymbol{k}_{\pm}^{(t)}\rangle) = 1 - O\Big(\frac{N \cdot \mathrm{SNR}^2 \cdot \big(\log(6N^2M^2/\delta)\big)^3}{d_h^{\frac{1}{2}}}\Big) \tag{111}$$

Substituting $t = T = \frac{C_{19}}{\eta \epsilon \|\boldsymbol{\mu}\|_2^2 \|\boldsymbol{w}_O\|_2^2}$ into propositions $\mathcal{F}(t)$ and get

$$\begin{aligned} V_+^{(t)} &\geq \log\Big(\exp(V_+^{(T_2)}) + \eta C_{17}\|\boldsymbol{\mu}\|_2^2 \|\boldsymbol{w}_O\|_2^2 (t - T_2)\Big) \\ &\geq \log\Big(\exp(V_+^{(T_2)}) + \frac{C_{20}}{\epsilon}\Big) \\ &\geq \log\Big(\frac{C_{20}}{\epsilon}\Big), \end{aligned} \tag{112}$$

$$|V_{n,i}^{(t)}| = O(1). \tag{113}$$

For $n \in S_+$, we bound $f(\boldsymbol{X}_n, \theta(t))$ as follows

$$f(\boldsymbol{X}_n, \theta(t)) = \frac{1}{M} \sum_{l=1}^{M} \boldsymbol{\varphi}(\boldsymbol{x}_{n,l}^\top \boldsymbol{W}_Q \boldsymbol{W}_K^\top (\boldsymbol{X}_n)^\top) \boldsymbol{X}_n \boldsymbol{W}_V \boldsymbol{w}_O$$

$$\frac{1}{M} \cdot \Bigg( \Big( \frac{\exp(\langle \boldsymbol{q}_+^{(t)}, \boldsymbol{k}_+^{(t)} \rangle)}{\exp(\langle \boldsymbol{q}_+^{(t)}, \boldsymbol{k}_+^{(t)} \rangle) + \sum\limits_{j=2}^{M} \exp(\langle \boldsymbol{q}_+^{(t)}, \boldsymbol{k}_{n,j}^{(t)} \rangle)}$$

$$+ \sum_{i=2}^{M} \frac{\exp(\langle \boldsymbol{q}_{n,i}^{(t)}, \boldsymbol{k}_+^{(t)} \rangle)}{\exp(\langle \boldsymbol{q}_{n,i}^{(t)}, \boldsymbol{k}_+^{(t)} \rangle) + \sum\limits_{j=2}^{M} \exp(\langle \boldsymbol{q}_{n,i}^{(t)}, \boldsymbol{k}_{n,j}^{(t)} \rangle)} \Big) \cdot V_+^{(T)}$$

$$+ \sum_{i=2}^{M} \Big( \frac{\exp(\langle \boldsymbol{q}_+^{(t)}, \boldsymbol{k}_{n,i}^{(t)} \rangle)}{\exp(\langle \boldsymbol{q}_+^{(t)}, \boldsymbol{k}_+^{(t)} \rangle) + \sum\limits_{j=2}^{M} \exp(\langle \boldsymbol{q}_+^{(t)}, \boldsymbol{k}_{n,j}^{(t)} \rangle)}$$

$$+ \sum_{l=2}^{M} \frac{\exp(\langle \boldsymbol{q}_{n,l}^{(t)}, \boldsymbol{k}_{n,i}^{(t)} \rangle)}{\exp(\langle \boldsymbol{q}_{n,l}^{(t)}, \boldsymbol{k}_+^{(t)} \rangle) + \sum\limits_{j=2}^{M} \exp(\langle \boldsymbol{q}_{n,l}^{(t)}, \boldsymbol{k}_{n,j}^{(t)} \rangle)} \Big) \cdot V_{n,i}^{(t)} \Bigg)$$

$$\geq \frac{1}{M} \cdot \Big( 1 - O\Big( \frac{N \big( \log(6N^2 M^2 / \delta) \big)^3}{d_h^{\frac{1}{2}}} \Big)$$

$$+ (M-1) \cdot \Big( 1 - O\Big( \frac{N \cdot \mathrm{SNR}^2 \cdot \big( \log(6N^2 M^2 / \delta) \big)^3}{d_h^{\frac{1}{2}}} \Big) \Big) \Big) \cdot \log\Big( \frac{C_{20}}{\epsilon} \Big)$$

$$- \frac{1}{M} \cdot (M-1) \cdot \Big( O\Big( \frac{N \big( \log(6N^2 M^2 / \delta) \big)^3}{d_h^{\frac{1}{2}}} \Big)$$

$$+ (M-1) \cdot O\Big( \frac{N \cdot \mathrm{SNR}^2 \cdot \big( \log(6N^2 M^2 / \delta) \big)^3}{d_h^{\frac{1}{2}}} \Big) \Big) \cdot O(1)$$

$$= \log\Big( \frac{C_{20}}{\epsilon} \Big) - O\Big( \frac{N \cdot \mathrm{SNR}^2 \cdot \big( \log(6N^2 M^2 / \delta) \big)^3 \log\big( \frac{C_{20}}{\epsilon} \big)}{d_h^{\frac{1}{2}}} \Big)$$

$$\geq \log\Big( \frac{C_{20}}{\epsilon} \Big) - \log(C_{20})$$

$$\geq \log\Big( \frac{1}{\epsilon} \Big),$$

where the first inequality is by plugging (108), (109), (110) and (111). For the second inequality, by $d_h = \widetilde{\Omega}\Big( \max\{\mathrm{SNR}^4, \mathrm{SNR}^{-4}\} N^2 \epsilon^{-2} \Big)$, we have $O\Big( \frac{N \cdot \mathrm{SNR}^2 \cdot \big( \log(6N^2 M^2 / \delta) \big)^3 \log\big( \frac{C_{20}}{\epsilon} \big)}{d_h^{\frac{1}{2}}} \Big) = o(1) = \log(1 + o(1)) \leq \log(C_{20})$ as long as $C_{20}$ is sufficiently large. Then we have

$$\begin{aligned}
\ell_n^{(t)} &= \log \Big( 1 + \exp(-f(\boldsymbol{X}_n, \theta(t))) \Big) \\
&\leq \exp(-f(\boldsymbol{X}_n, \theta(t))) \\
&\leq \exp\Big( -\log\Big( \frac{1}{\epsilon} \Big) \Big) \\
&\leq \epsilon.
\end{aligned} \tag{115}$$

Similarly, we have $\ell_n^{(t)} \leq \epsilon$ for $n \in S_-$. Therefore, we have $L_S(\theta(T)) = \frac{1}{N} \sum\limits_{n=1}^{N} \ell_n^{(t)} \leq \epsilon$.

## D.4 Population Loss

Consider a new data point $(\boldsymbol{X}, y)$ generated from the distribution defined in Definition 3.1. Without loss of generality, we suppose that the signal token is $\boldsymbol{\mu}_+$ and the label is 1, i.e. $\boldsymbol{X} = (\boldsymbol{\mu}_+, \boldsymbol{\xi}_2, \ldots, \boldsymbol{\xi}_M)$, $y = 1$. We will first give the bounds of attention, then the bounds of the output of ViT, and finally the bound of the expectation of loss.

**Lemma D.8** (Bound of Attention). *Under the same conditions as Theorem 4.1, with probability at least $1 - \delta/2N^2 M$, we have that*

$$\frac{\exp(\boldsymbol{\mu}_+^\top \boldsymbol{W}_Q^{(T_3)} \boldsymbol{W}_K^{(T_3)\top} \boldsymbol{\mu}_+)}{\exp(\boldsymbol{\mu}_+^\top \boldsymbol{W}_Q^{(T_3)} \boldsymbol{W}_K^{(T_3)\top} \boldsymbol{\mu}_+) + \sum\limits_{i=2}^{M} \exp(\boldsymbol{\mu}_+^\top \boldsymbol{W}_Q^{(T_3)} \boldsymbol{W}_K^{(T_3)\top} \boldsymbol{\xi}_i)} \geq \frac{1}{2M}.$$

*Proof of Lemma D.8.* By the result of D.2 and D.3, $\boldsymbol{\mu}_+^\top \boldsymbol{W}_Q^{(T_1)} \boldsymbol{W}_K^{(T_1)\top} \boldsymbol{\mu}_+ = o(1)$, and $\boldsymbol{\mu}_+^\top \boldsymbol{W}_Q^{(t)} \boldsymbol{W}_K^{(t)\top} \boldsymbol{\mu}_+$ is monotonically non-decreasing for $t \in [T_1, T_3]$, we have $\boldsymbol{\mu}_+^\top \boldsymbol{W}_Q^{(T_3)} \boldsymbol{W}_K^{(T_3)\top} \boldsymbol{\mu}_+ \geq -o(1)$.

By proposition $\mathcal{G}(t)$ in D.3, we have $\|\boldsymbol{\mu}_+^\top \boldsymbol{W}_Q^{(T_3)}\|_2^2 = \Theta\big(\|\boldsymbol{\mu}\|_2^2 \sigma_h^2 d_h\big)$. In order to give bounds for $\boldsymbol{\mu}_+^\top \boldsymbol{W}_Q^{(T_3)} \boldsymbol{W}_K^{(T_3)\top} \boldsymbol{\xi}_i$, we provide a bound for the F-norm of $\boldsymbol{W}_K^{(T_3)} - \boldsymbol{W}_K^{(0)}$. According to the properties of F-norm, we have

$$\|\boldsymbol{W}_K^{(T_3)} - \boldsymbol{W}_K^{(0)}\|_F \leq \|\boldsymbol{W}_K^{(T_1)} - \boldsymbol{W}_K^{(0)}\|_F + \|\boldsymbol{W}_K^{(T_2)} - \boldsymbol{W}_K^{(T_1)}\|_F + \|\boldsymbol{W}_K^{(T_3)} - \boldsymbol{W}_K^{(T_2)}\|_F \quad (116)$$

Next, we will provide the bounds of $\|\boldsymbol{W}_K^{(T_1)} - \boldsymbol{W}_K^{(0)}\|_F$, $\|\boldsymbol{W}_K^{(T_2)} - \boldsymbol{W}_K^{(T_1)}\|_F$, and $\|\boldsymbol{W}_K^{(T_3)} - \boldsymbol{W}_K^{(T_2)}\|_F$ respectively.

**Bounding $\|\boldsymbol{W}_K^{(T_1)} - \boldsymbol{W}_K^{(0)}\|_F$:** Recall that

$$\nabla_{\boldsymbol{W}_K} L_S(\theta) = \frac{1}{NM} \sum_{n \in S_+} \ell_n'(\theta)(\boldsymbol{X}_n^\top (diag(\boldsymbol{\varphi}_{n,1}) - \boldsymbol{\varphi}_{n,1}^\top \boldsymbol{\varphi}_{n,1}) \boldsymbol{X}_n \boldsymbol{W}_V \boldsymbol{w}_O \boldsymbol{\mu}_+^\top$$

$$+ \sum_{i=2}^{M} \boldsymbol{X}_n^\top (diag(\boldsymbol{\varphi}_{n,i}) - \boldsymbol{\varphi}_{n,i}^\top \boldsymbol{\varphi}_{n,i}) \boldsymbol{X}_n \boldsymbol{W}_V \boldsymbol{w}_O \boldsymbol{\xi}_{n,i}^\top) \boldsymbol{W}_Q$$

$$- \frac{1}{NM} \sum_{n \in S_-} \ell_n'(\theta)(\boldsymbol{X}_n^\top (diag(\boldsymbol{\varphi}_{n,1}) - \boldsymbol{\varphi}_{n,1}^\top \boldsymbol{\varphi}_{n,1}) \boldsymbol{X}_n \boldsymbol{W}_V \boldsymbol{w}_O \boldsymbol{\mu}_-^\top$$

$$+ \sum_{i=2}^{M} \boldsymbol{X}_n^\top (diag(\boldsymbol{\varphi}_{n,i}) - \boldsymbol{\varphi}_{n,i}^\top \boldsymbol{\varphi}_{n,i}) \boldsymbol{X}_n \boldsymbol{W}_V \boldsymbol{w}_O \boldsymbol{\xi}_{n,i}^\top) \boldsymbol{W}_Q,$$

so we have

$$\|\boldsymbol{W}_K^{(t+1)} - \boldsymbol{W}_K^{(t)}\|_F = \|\eta \nabla_{\boldsymbol{W}_K} L_S(\theta(t))\|_F$$

$$\leq \frac{\eta}{NM} \sum_{n \in S_+} |\ell_n'^{(t)}|(\|\boldsymbol{X}_n^{(t)\top}\|_F \|(diag(\boldsymbol{\varphi}_{n,1}^{(t)}) - \boldsymbol{\varphi}_{n,1}^{(t)\top} \boldsymbol{\varphi}_{n,1}^{(t)})\|_F \|\boldsymbol{X}_n \boldsymbol{W}_V^{(t)} \boldsymbol{w}_O\|_F \|\boldsymbol{\mu}_+^\top \boldsymbol{W}_Q^{(t)}\|_F$$

$$+ \sum_{i=2}^{M} \|\boldsymbol{X}_n^\top\|_F \|(diag(\boldsymbol{\varphi}_{n,i}^{(t)}) - \boldsymbol{\varphi}_{n,i}^{(t)\top} \boldsymbol{\varphi}_{n,i}^{(t)})\|_F \|\boldsymbol{X}_n \boldsymbol{W}_V^{(t)} \boldsymbol{w}_O\|_F \|\boldsymbol{\xi}_{n,i}^\top \boldsymbol{W}_Q^{(t)}\|_F)$$

$$+ \frac{\eta}{NM} \sum_{n \in S_-} |\ell_n'^{(t)}|(\|\boldsymbol{X}_n^\top\|_F \|(diag(\boldsymbol{\varphi}_{n,1}^{(t)}) - \boldsymbol{\varphi}_{n,1}^{(t)\top} \boldsymbol{\varphi}_{n,1}^{(t)})\|_F \|\boldsymbol{X}_n \boldsymbol{W}_V^{(t)} \boldsymbol{w}_O\|_F \|\boldsymbol{\mu}_-^\top \boldsymbol{W}_Q^{(t)}\|_F$$

$$+ \sum_{i=2}^{M} \|\boldsymbol{X}_n^\top\|_F \|(diag(\boldsymbol{\varphi}_{n,i}^{(t)}) - \boldsymbol{\varphi}_{n,i}^{(t)\top} \boldsymbol{\varphi}_{n,i}^{(t)})\|_F \|\boldsymbol{X}_n \boldsymbol{W}_V^{(t)} \boldsymbol{w}_O\|_F \|\boldsymbol{\xi}_{n,i}^\top \boldsymbol{W}_Q^{(t)}\|_F)$$

$$(117)$$

We bound $|\ell_n'^{(t)}|$, $\|\boldsymbol{X}_n\|_F$, $\|(diag(\boldsymbol{\varphi}_{n,i}^{(t)}) - \boldsymbol{\varphi}_{n,i}^{(t)\top} \boldsymbol{\varphi}_{n,i}^{(t)})\|_F$, $\|\boldsymbol{X}_n \boldsymbol{W}_V^{(t)} \boldsymbol{w}_O\|_F$, $\|\boldsymbol{\mu}_\pm^\top \boldsymbol{W}_Q^{(t)}\|_F$ and $\|\boldsymbol{\xi}_{n,i}^\top \boldsymbol{W}_Q^{(t)}\|_F$ respectively.

- $|\ell_n'^{(t)}| \le 1$

- By Lemma C.4, $\|\boldsymbol{X}_n\|_F \le \sqrt{\|\boldsymbol{\mu}\|_2^2 + (M-1)\frac{3\sigma_p^2 d}{2}} = O\big(\max\{\|\boldsymbol{\mu}\|_2, \sigma_p\sqrt{d}\}\big)$

- Note that each element of vector $\boldsymbol{\varphi}_{n,i}$ has an upper bound 1, and the number of tokens has a constant upper bound ($M = \Theta(1)$), then we have $\|(diag(\boldsymbol{\varphi}_{n,i}^{(t)}) - \boldsymbol{\varphi}_{n,i}^{(t)\top}\boldsymbol{\varphi}_{n,i}^{(t)})\|_F = O(1)$.

- By Lemma D.3, we have $|V_+^{(t)}|, |V_-^{(t)}|, |V_{n,i}^{(t)}| = O(d_h^{-\frac{1}{4}})$, so we have

$$\|\boldsymbol{X}_n\boldsymbol{W}_V\boldsymbol{w}_O\|_F = \sqrt{\big(V_+^{(t)}\big)^2 + \sum_{i=2}^M \big(V_{n,i}^{(t)}\big)^2} = \sqrt{M \cdot \big(d_h^{-\frac{1}{4}}\big)^2} = O\big(d_h^{-\frac{1}{4}}\big)$$

- By Lemma D.4, we have $\|\boldsymbol{\mu}_\pm^\top \boldsymbol{W}_Q^{(t)}\|_F = \Theta\big(\|\boldsymbol{\mu}\|_2\sigma_h d_h^{\frac{1}{2}}\big)$, $\|\boldsymbol{\xi}_{n,i}^\top \boldsymbol{W}_Q^{(t)}\|_F = \Theta\big(\sigma_p\sigma_h d^{\frac{1}{2}} d_h^{\frac{1}{2}}\big)$

Plugging them into (117) and get

$$\|\boldsymbol{W}_K^{(t+1)} - \boldsymbol{W}_K^{(t)}\|_F \le \frac{\eta}{NM} \cdot NM \cdot O\big(\max\{\|\boldsymbol{\mu}\|_2, \sigma_p\sqrt{d}\}\big) \cdot O\big(d_h^{-\frac{1}{4}}\big) \cdot O\big(\max\{\|\boldsymbol{\mu}\|_2, \sigma_p\sqrt{d}\}\sigma_h d_h^{\frac{1}{2}}\big)$$
$$= O\big(\eta \cdot \max\{\|\boldsymbol{\mu}\|_2^2, \sigma_p^2 d\} \cdot \sigma_h d_h^{\frac{1}{4}}\big)$$
(118)

Taking a summation we have

$$\|\boldsymbol{W}_K^{(T_1)} - \boldsymbol{W}_K^{(0)}\|_F \le \sum_{s=0}^{T_1-1} \|\boldsymbol{W}_K^{(t+1)} - \boldsymbol{W}_K^{(t)}\|_F$$
$$\le O\Big(\frac{1}{\eta d_h^{\frac{1}{4}}\|\boldsymbol{\mu}\|_2^2\|\boldsymbol{w}_O\|_2^2}\Big) \cdot O\big(\eta \cdot \max\{\|\boldsymbol{\mu}\|_2^2, \sigma_p^2 d\} \cdot \sigma_h d_h^{\frac{1}{4}}\big)$$
$$= O\Big(\frac{\max\{\|\boldsymbol{\mu}\|_2^2, \sigma_p^2 d\}}{\|\boldsymbol{\mu}\|_2^2} \cdot \sigma_h\Big)$$
$$\le O\big(N\sigma_h\big),$$
(119)

where the last inequality is by $N \cdot \mathrm{SNR}^2 = \Omega(1)$.

**Bounding** $\|\boldsymbol{W}_K^{(T_2)} - \boldsymbol{W}_K^{(T_1)}\|_F$: By F.7, we know that each element of matrix $(diag(\boldsymbol{\varphi}_{n,1}^{(t)}) - \boldsymbol{\varphi}_{n,1}^{(t)\top}\boldsymbol{\varphi}_{n,1}^{(t)})$ have upper bound $O\Big(\frac{1}{1+\frac{\eta^2\|\boldsymbol{\mu}\|_2^4\|\boldsymbol{w}_O\|_2^2 d_h^{\frac{1}{2}}}{N\big(\log(6N^2M^2/\delta)\big)^2}\cdot(t-T_1)(t-T_1-1)}\Big)$, and each element of matrix $(diag(\boldsymbol{\varphi}_{n,i}^{(t)}) - \boldsymbol{\varphi}_{n,i}^{(t)\top}\boldsymbol{\varphi}_{n,i}^{(t)})$ have upper bound $O\Big(\frac{1}{1+\frac{\eta^2\sigma_p^2 d\|\boldsymbol{\mu}\|_2^2\|\boldsymbol{w}_O\|_2^2 d_h^{\frac{1}{2}}}{N\big(\log(6N^2M^2/\delta)\big)^2}\cdot(t-T_1)(t-T_1-1)}\Big)$. Considering that the number of tokens has a constant upper bound ($M = \Theta(1)$), we have

$$\|(diag(\boldsymbol{\varphi}_{n,1}^{(t)}) - \boldsymbol{\varphi}_{n,1}^{(t)\top}\boldsymbol{\varphi}_{n,1}^{(t)})\|_F = O\Big(\frac{1}{1 + \frac{\eta^2\|\boldsymbol{\mu}\|_2^4\|\boldsymbol{w}_O\|_2^2 d_h^{\frac{1}{2}}}{N\big(\log(6N^2M^2/\delta)\big)^2} \cdot (t-T_1)(t-T_1-1)}\Big) \quad (120)$$

$$\|(diag(\boldsymbol{\varphi}_{n,i}^{(t)}) - \boldsymbol{\varphi}_{n,i}^{(t)\top}\boldsymbol{\varphi}_{n,i}^{(t)})\|_F = O\Big(\frac{1}{1 + \frac{\eta^2\sigma_p^2 d\|\boldsymbol{\mu}\|_2^2\|\boldsymbol{w}_O\|_2^2 d_h^{\frac{1}{2}}}{N\big(\log(6N^2M^2/\delta)\big)^2} \cdot (t-T_1)(t-T_1-1)}\Big) \quad (121)$$

By $\mathcal{B}(t)$ in D.2, we have $|V_\pm^{(t)}|, |V_{n,i}^{(t)}| \le O(d_h^{-\frac{1}{4}}) + \eta C_4\|\boldsymbol{\mu}\|_2^2\|\boldsymbol{w}_O\|_2^2(t-T_1)$ for $t \in [T_1, T_2]$. We further have

$$\|\boldsymbol{X}_n\boldsymbol{W}_V^{(t)}\boldsymbol{w}_O\|_F = O\Big(d_h^{-\frac{1}{4}} + \eta\|\boldsymbol{\mu}\|_2^2\|\boldsymbol{w}_O\|_2^2(t-T_1)\Big).$$

Then we have

$$\|(diag(\boldsymbol{\varphi}_{n,1}^{(t)}) - \boldsymbol{\varphi}_{n,1}^{(t)\top}\boldsymbol{\varphi}_{n,1}^{(t)})\|_F \cdot \|\boldsymbol{X}_n \boldsymbol{W}_V^{(t)}\boldsymbol{w}_O\|_F$$

$$= O\Big(\frac{1}{1 + \frac{\eta^2\|\boldsymbol{\mu}\|_2^4\|\boldsymbol{w}_O\|_2^2 d_h^{\frac{1}{2}}}{N\big(\log(6N^2M^2/\delta)\big)^2} \cdot (t - T_1)(t - T_1 - 1)}\Big) \cdot O\Big(d_h^{-\frac{1}{4}} + \eta\|\boldsymbol{\mu}\|_2^2\|\boldsymbol{w}_O\|_2^2(t - T_1)\Big)$$

$$= O\Big(N^{\frac{1}{2}}d_h^{-\frac{1}{4}}\log(6N^2M^2/\delta)\Big), \tag{122}$$

The reason for the last equation is similar to (256). We also have

$$\|(diag(\boldsymbol{\varphi}_{n,i}^{(t)}) - \boldsymbol{\varphi}_{n,i}^{(t)\top}\boldsymbol{\varphi}_{n,i}^{(t)})\|_F \cdot \|\boldsymbol{X}_n \boldsymbol{W}_V^{(t)}\boldsymbol{w}_O\|_F$$

$$= O\Big(\frac{1}{1 + \frac{\eta^2\sigma_p^2 d\|\boldsymbol{\mu}\|_2^2\|\boldsymbol{w}_O\|_2^2 d_h^{\frac{1}{2}}}{N\big(\log(6N^2M^2/\delta)\big)^2} \cdot (t - T_1)(t - T_1 - 1)}\Big) \cdot O\Big(d_h^{-\frac{1}{4}} + \eta\|\boldsymbol{\mu}\|_2^2\|\boldsymbol{w}_O\|_2^2(t - T_1)\Big)$$

$$= O\Big(\frac{\|\boldsymbol{\mu}\|_2}{\sigma_p\sqrt{d}}N^{\frac{1}{2}}d_h^{-\frac{1}{4}}\log(6N^2M^2/\delta)\Big). \tag{123}$$

Plugging them into (117) and get

$$\|\boldsymbol{W}_K^{(t+1)} - \boldsymbol{W}_K^{(t)}\|_F \le \frac{\eta}{NM} \cdot N \cdot O\big(\max\{\|\boldsymbol{\mu}\|_2, \sigma_p\sqrt{d}\}\big) \cdot \Big(O\Big(N^{\frac{1}{2}}d_h^{-\frac{1}{4}}\log(6N^2M^2/\delta)\Big) \cdot O\big(\|\boldsymbol{\mu}\|_2\sigma_h d_h^{\frac{1}{2}}\big)$$

$$+ (M - 1) \cdot O\Big(\frac{\|\boldsymbol{\mu}\|_2}{\sigma_p\sqrt{d}}N^{\frac{1}{2}}d_h^{-\frac{1}{4}}\log(6N^2M^2/\delta)\Big) \cdot O\big(\sigma_p\sqrt{d}\sigma_h d_h^{\frac{1}{2}}\big)\Big)$$

$$= O\Big(\eta \cdot \max\{\|\boldsymbol{\mu}\|_2, \sigma_p\sqrt{d}\}N^{\frac{1}{2}}\|\boldsymbol{\mu}\|_2\sigma_h d_h^{\frac{1}{4}}\log(6N^2M^2/\delta)\Big) \tag{124}$$

Taking a summation we have

$$\|\boldsymbol{W}_K^{(T_2)} - \boldsymbol{W}_K^{(T_1)}\|_F \le \sum_{s=T_1}^{T_2-1} \|\boldsymbol{W}_K^{(t+1)} - \boldsymbol{W}_K^{(t)}\|_F$$

$$\le O\big(\frac{1}{\eta\|\boldsymbol{\mu}\|_2^2\|\boldsymbol{w}_O\|_2^2}\big) \cdot O\Big(\eta\max\{\|\boldsymbol{\mu}\|_2, \sigma_p\sqrt{d}\}N^{\frac{1}{2}}\|\boldsymbol{\mu}\|_2\sigma_h d_h^{\frac{1}{4}}\log(6N^2M^2/\delta)\Big)$$

$$= O\Big(\frac{\max\{\|\boldsymbol{\mu}\|_2, \sigma_p\sqrt{d}\}}{\|\boldsymbol{\mu}\|_2} \cdot N^{\frac{1}{2}}\sigma_h d_h^{\frac{1}{4}}\log(6N^2M^2/\delta)\Big)$$

$$\le O\Big(N^{\frac{1}{2}}(1 + \mathrm{SNR}^{-1})\sigma_h d_h^{\frac{1}{4}}\log(6N^2M^2/\delta)\Big), \tag{125}$$

**Bounding $\|\boldsymbol{W}_K^{(T_3)} - \boldsymbol{W}_K^{(T_2)}\|_F$:** Recall that ((64), (65))

$$\frac{\exp(\langle \boldsymbol{q}_\pm^{(t)}, \boldsymbol{k}_{n,j}^{(t)}\rangle)}{\exp(\langle \boldsymbol{q}_\pm^{(t)}, \boldsymbol{k}_\pm^{(t)}\rangle) + \sum_{j'=2}^{M}\exp(\langle \boldsymbol{q}_\pm^{(t)}, \boldsymbol{k}_{n,j'}^{(t)}\rangle)} = O\Big(\frac{N\big(\log(6N^2M^2/\delta)\big)^3}{d_h^{\frac{1}{2}}}\Big),$$

$$\frac{\exp(\langle \boldsymbol{q}_{n,i}^{(t)}, \boldsymbol{k}_{n,j}^{(t)}\rangle)}{\exp(\langle \boldsymbol{q}_{n,i}^{(t)}, \boldsymbol{k}_+^{(t)}\rangle) + \sum_{j'=2}^{M}\exp(\langle \boldsymbol{q}_{n,i}^{(t)}, \boldsymbol{k}_{n,j'}^{(t)}\rangle)} = O\Big(\frac{N\|\boldsymbol{\mu}\|_2^2\big(\log(6N^2M^2/\delta)\big)^3}{\sigma_p^2 dd_h^{\frac{1}{2}}}\Big)$$

for $i, j \in [M]\backslash\{1\}, n \in S_\pm, t \in [T_2, T_3]$. We have

$$\|(diag(\boldsymbol{\varphi}_{n,1}^{(t)}) - \boldsymbol{\varphi}_{n,1}^{(t)\top}\boldsymbol{\varphi}_{n,1}^{(t)})\|_F = O\Big(\frac{N\big(\log(6N^2M^2/\delta)\big)^3}{d_h^{\frac{1}{2}}}\Big), \tag{126}$$

$$\|(diag(\boldsymbol{\varphi}_{n,i}^{(t)}) - \boldsymbol{\varphi}_{n,i}^{(t)\top}\boldsymbol{\varphi}_{n,i}^{(t)})\|_F = O\Big(\frac{N\|\boldsymbol{\mu}\|_2^2\big(\log(6N^2M^2/\delta)\big)^3}{\sigma_p^2 dd_h^{\frac{1}{2}}}\Big). \tag{127}$$

By $\mathcal{F}(t)$ in D.3, we have

$$|V_{\pm}^{(t)}| \le 2\log\big(O(\frac{1}{\epsilon})\big)$$

$$|V_{n,i}^{(t)}| = O(1)$$

we further have

$$\|\boldsymbol{X}_n\boldsymbol{W}_V^{(t)}\boldsymbol{w}_O\|_F = O\Big(\log O(\frac{1}{\epsilon})\Big).$$

Plugging them into (117) and get

$$\|\boldsymbol{W}_K^{(t+1)} - \boldsymbol{W}_K^{(t)}\|_F \le \frac{\eta}{NM}\cdot N\cdot O\big(\max\{\|\boldsymbol{\mu}\|_2, \sigma_p\sqrt{d}\}\big)\cdot\Big(O\Big(\frac{N\big(\log(6N^2M^2/\delta)\big)^3}{d_h^{\frac{1}{2}}}\Big)\cdot O\big(\|\boldsymbol{\mu}\|_2\sigma_h d_h^{\frac{1}{2}}\big)$$

$$+ (M-1)\cdot O\Big(\frac{N\|\boldsymbol{\mu}\|_2^2\big(\log(6N^2M^2/\delta)\big)^3}{\sigma_p^2 dd_h^{\frac{1}{2}}}\Big)\cdot O\big(\sigma_p\sqrt{d}\sigma_h d_h^{\frac{1}{2}}\big)\Big)\cdot O\Big(\log\big(O(\frac{1}{\epsilon})\big)\Big)$$

$$= O\Big(\eta\cdot\max\{\|\boldsymbol{\mu}\|_2^2, \sigma_p^2 d\}\cdot\frac{N\|\boldsymbol{\mu}\|_2\big(\log(6N^2M^2/\delta)\big)^3\log\big(O(\frac{1}{\epsilon})\big)\sigma_h}{\sigma_p d^{\frac{1}{2}}}\Big) \tag{128}$$

Taking a summation we have

$$\|\boldsymbol{W}_K^{(T_3)} - \boldsymbol{W}_K^{(T_2)}\|_F \le \sum_{s=T_2}^{T_3-1}\|\boldsymbol{W}_K^{(t+1)} - \boldsymbol{W}_K^{(t)}\|_F$$

$$\le O\Big(\frac{1}{\eta\epsilon\|\boldsymbol{\mu}\|_2^2\|\boldsymbol{w}_O\|_2^2}\Big)\cdot O\Big(\eta\cdot\max\{\|\boldsymbol{\mu}\|_2^2, \sigma_p^2 d\}\cdot\frac{N\|\boldsymbol{\mu}\|_2\big(\log(6N^2M^2/\delta)\big)^3\log\big(O(\frac{1}{\epsilon})\big)\sigma_h d_h^{\frac{1}{2}}}{\sigma_p d^{\frac{1}{2}}d_h^{\frac{1}{2}}}\Big)$$

$$= O\Big(\max\{\|\boldsymbol{\mu}\|_2^2, \sigma_p^2 d\}\cdot\frac{N\big(\log(6N^2M^2/\delta)\big)^3\log\big(O(\frac{1}{\epsilon})\big)\sigma_h}{\epsilon\|\boldsymbol{\mu}\|_2\sigma_p d^{\frac{1}{2}}}\Big). \tag{129}$$

Combining (116), (119), (125) and (129) and get

$$\|\boldsymbol{W}_K^{(T_3)} - \boldsymbol{W}_K^{(0)}\|_F \le \|\boldsymbol{W}_K^{(T_1)} - \boldsymbol{W}_K^{(0)}\|_F + \|\boldsymbol{W}_K^{(T_2)} - \boldsymbol{W}_K^{(T_1)}\|_F + \|\boldsymbol{W}_K^{(T_3)} - \boldsymbol{W}_K^{(T_2)}\|_F$$

$$= O\Big(N\sigma_h\Big) + O\Big(N^{\frac{1}{2}}(1 + \text{SNR}^{-1})\sigma_h d_h^{\frac{1}{4}}\log(6N^2M^2/\delta)\Big)$$

$$+ O\Big(\max\{\|\boldsymbol{\mu}\|_2^2, \sigma_p^2 d\}\cdot\frac{N\big(\log(6N^2M^2/\delta)\big)^3\log\big(O(\frac{1}{\epsilon})\big)\sigma_h}{\epsilon\|\boldsymbol{\mu}\|_2\sigma_p d^{\frac{1}{2}}}\Big)$$

$$= O\Big(\sigma_h d_h^{\frac{1}{2}}\Big), \tag{130}$$

where the last equality is by $d_h = \widetilde{\Omega}\Big(\max\{\text{SNR}^4, \text{SNR}^{-4}\}N^2\epsilon^{-2}\Big)$.

**Bounding $\boldsymbol{\mu}_+^\top\boldsymbol{W}_Q^{(T_3)}\boldsymbol{W}_K^{(T_3)\top}\boldsymbol{\xi}_i$:** We decompose $\boldsymbol{W}_K^{(T_3)}$ into $\boldsymbol{W}_K^{(0)} + \Big(\boldsymbol{W}_K^{(T_3)} - \boldsymbol{W}_K^{(0)}\Big)$, then we have

$$\boldsymbol{\mu}_+^\top\boldsymbol{W}_Q^{(T_3)}\boldsymbol{W}_K^{(T_3)\top}\boldsymbol{\xi}_i = \boldsymbol{\mu}_+^\top\boldsymbol{W}_Q^{(T_3)}\boldsymbol{W}_K^{(0)\top}\boldsymbol{\xi}_i + \boldsymbol{\mu}_+^\top\boldsymbol{W}_Q^{(T_3)}\Big(\boldsymbol{W}_K^{(T_3)\top} - \boldsymbol{W}_K^{(0)\top}\Big)\boldsymbol{\xi}_i$$

Now we bound $\boldsymbol{\mu}_+^\top\boldsymbol{W}_Q^{(T_3)}\boldsymbol{W}_K^{(0)\top}\boldsymbol{\xi}_i$ and $\boldsymbol{\mu}_+^\top\boldsymbol{W}_Q^{(T_3)}\Big(\boldsymbol{W}_K^{(T_3)\top} - \boldsymbol{W}_K^{(0)\top}\Big)\boldsymbol{\xi}_i$ respectively.

**Bounding $\boldsymbol{\mu}_+^\top\boldsymbol{W}_Q^{(T_3)}\boldsymbol{W}_K^{(0)\top}\boldsymbol{\xi}_i$:** Note that $\|\boldsymbol{\mu}_+^\top\boldsymbol{W}_Q^{(T_3)}\|_2^2 = \Theta\big(\|\boldsymbol{\mu}\|_2^2\sigma_h^2 d_h\big)$ and each element of $\boldsymbol{W}_K^{(0)\top}$ is sampled from a Gaussian distribution $\mathcal{N}(0, \sigma_h^2)$, we have that each element of

$\boldsymbol{\mu}_+^\top \boldsymbol{W}_Q^{(T_3)} \boldsymbol{W}_K^{(0)\top}$ is a random variable with mean zero and standard deviation smaller than $C_{21}\|\boldsymbol{\mu}\|_2 \sigma_h^2 d_h^{\frac{1}{2}}$. Therefore, for any $i \in [M]\backslash\{1\}$ by Bernstein's inequality, with probability at least $1 - \delta/3N^2M^2$,

$$\boldsymbol{\mu}_+^\top \boldsymbol{W}_Q^{(T_3)} \boldsymbol{W}_K^{(0)\top} \boldsymbol{\xi}_i = O(\|\boldsymbol{\mu}\|_2 \sigma_p \sigma_h^2 d_h^{\frac{1}{2}} \sqrt{d \log(6N^2M^2/\delta)}) = o(1),$$

where the last equality is by $\sigma_h^2 \le \min\{\|\boldsymbol{\mu}\|_2^{-2}, (\sigma_p^2 d)^{-1}\} \cdot d_h^{-\frac{1}{2}} \cdot \left(\log(6N^2M^2/\delta)\right)^{-\frac{3}{2}}$.

**Bounding $\boldsymbol{\mu}_+^\top \boldsymbol{W}_Q^{(T_3)} \left(\boldsymbol{W}_K^{(T_3)\top} - \boldsymbol{W}_K^{(0)\top}\right) \boldsymbol{\xi}_i$:** By (130), we have

$$\begin{aligned} \left\|\boldsymbol{\mu}_+^\top \boldsymbol{W}_Q^{(T_3)} \left(\boldsymbol{W}_K^{(T_3)\top} - \boldsymbol{W}_K^{(0)\top}\right)\right\|_2 &\le \|\boldsymbol{\mu}_+^\top \boldsymbol{W}_Q^{(T_3)}\|_2 \|\boldsymbol{W}_K^{(T_3)\top} - \boldsymbol{W}_K^{(0)\top}\|_F \\ &\le C_{22}\|\boldsymbol{\mu}\|_2 \sigma_h^2 d_h. \end{aligned} \tag{131}$$

Therefore $\boldsymbol{\mu}_+^\top \boldsymbol{W}_Q^{(T_3)} \left(\boldsymbol{W}_K^{(T_3)\top} - \boldsymbol{W}_K^{(0)\top}\right) \boldsymbol{\xi}_i$ is a Gaussian distribution with mean zero and standard deviation smaller than $C_{22}\|\boldsymbol{\mu}\|_2 \sigma_p \sigma_h^2 d_h$. By Gaussian tail bound, for any $i \in [M]\backslash\{1\}$, with probability at least $1 - \delta/6N^2M^2$,

$$\boldsymbol{\mu}_+^\top \boldsymbol{W}_Q^{(T_3)} \left(\boldsymbol{W}_K^{(T_3)\top} - \boldsymbol{W}_K^{(0)\top}\right) \boldsymbol{\xi}_i \le C_{22}\|\boldsymbol{\mu}\|_2 \sigma_p \sigma_h^2 d_h \sqrt{2\log\left(12N^2M^2/\delta\right)} = o(1),$$

where the last equality is by $d = \widetilde{\Omega}\left(\epsilon^{-2} N^2 d_h\right)$ and $\sigma_h^2 \le \min\{\|\boldsymbol{\mu}\|_2^{-2}, (\sigma_p^2 d)^{-1}\} \cdot d_h^{-\frac{1}{2}} \cdot \left(\log(6N^2M^2/\delta)\right)^{-\frac{3}{2}}$. Applying a union bound, with probability at least $1 - \delta/2N^2M$,

$$\begin{aligned} \boldsymbol{\mu}_+^\top \boldsymbol{W}_Q^{(T_3)} \boldsymbol{W}_K^{(T_3)\top} \boldsymbol{\xi}_i &= \boldsymbol{\mu}_+^\top \boldsymbol{W}_Q^{(T_3)} \boldsymbol{W}_K^{(0)\top} \boldsymbol{\xi}_i + \boldsymbol{\mu}_+^\top \boldsymbol{W}_Q^{(T_3)} \left(\boldsymbol{W}_K^{(T_3)\top} - \boldsymbol{W}_K^{(0)\top}\right) \boldsymbol{\xi}_i \\ &= o(1) + o(1) \\ &= o(1). \end{aligned}$$

Therefore, we have

$$\frac{\exp(\boldsymbol{\mu}_+^\top \boldsymbol{W}_Q^{(T_3)} \boldsymbol{W}_K^{(T_3)\top} \boldsymbol{\mu}_+)}{\exp(\boldsymbol{\mu}_+^\top \boldsymbol{W}_Q^{(T_3)} \boldsymbol{W}_K^{(T_3)\top} \boldsymbol{\mu}_+) + \sum_{i=2}^{M} \exp(\boldsymbol{\mu}_+^\top \boldsymbol{W}_Q^{(T_3)} \boldsymbol{W}_K^{(T_3)\top} \boldsymbol{\xi}_i)} \ge \frac{\exp(-o(1))}{\exp(-o(1)) + (M-1)\exp(o(1))}$$

$$\ge \frac{1}{1 + (M-1)\exp(o(1))}$$

$$\ge \frac{1}{1 + (M-1) + (M-1)o(1)}$$

$$\ge \frac{1}{M} - o(1)$$

$$\ge \frac{1}{2M},$$

which complete the proof.

**Lemma D.9** (Bound of V). *Under the same conditions as Theorem 4.1, with probability at least $1 - \delta/2N^2M$, we have that*

$$|\boldsymbol{\xi}_i^\top \boldsymbol{W}_V^{(T_3)} \boldsymbol{w}_O| \le 1$$

*Proof of Lemma D.9.* By (9), we have

$$\begin{aligned} \boldsymbol{W}_V^{(t+1)} \boldsymbol{w}_O - \boldsymbol{W}_V^{(t)} \boldsymbol{w}_O &= -\frac{\eta}{NM} \sum_{n=1}^{N} y_n \ell_n'^{(t)} [\boldsymbol{w}_O \sum_{l=1}^{M} \boldsymbol{\varphi}(\boldsymbol{x}_{n,l} \boldsymbol{W}_Q^{(t)} \boldsymbol{W}_K^{(t)\top} \boldsymbol{X}_n^\top) \boldsymbol{X}_n]^\top \boldsymbol{w}_O \\ &= -\frac{\eta}{NM} \sum_{n=1}^{N} y_n \ell_n'^{(t)} \sum_{l=1}^{M} \boldsymbol{X}_n^\top \boldsymbol{\varphi}(\boldsymbol{x}_{n,l} \boldsymbol{W}_Q^{(t)} \boldsymbol{W}_K^{(t)\top} \boldsymbol{X}_n^\top)^\top \|\boldsymbol{w}_O\|_2^2. \end{aligned} \tag{132}$$

Then we have

$$\|\boldsymbol{W}_V^{(t+1)}\boldsymbol{w}_O - \boldsymbol{W}_V^{(t)}\boldsymbol{w}_O\|_2 \leq \frac{\eta}{NM} \sum_{n=1}^{N} |y_n \ell_n'^{(t)}| \sum_{l=1}^{M} \|\boldsymbol{X}_n\|_F \|\boldsymbol{\varphi}(\boldsymbol{x}_{n,l}\boldsymbol{W}_Q^{(t)}\boldsymbol{W}_K^{(t)\top}\boldsymbol{X}_n^\top)\|_2 \|\boldsymbol{w}_O\|_2^2$$

$$\leq \frac{\eta}{NM} \cdot NM \cdot O\big(\max\{\|\boldsymbol{\mu}\|_2, \sigma_p\sqrt{d}\}\big) \cdot O(1) \cdot \|\boldsymbol{w}_O\|_2^2$$

$$= O\Big(\eta \cdot \max\{\|\boldsymbol{\mu}\|_2, \sigma_p\sqrt{d}\} \cdot \|\boldsymbol{w}_O\|_2^2\Big),$$
(133)

where the second inequality is by $|y_n \ell_n'^{(t)}| \leq 1$, $\|\boldsymbol{X}_n\|_F = O\big(\max\{\|\boldsymbol{\mu}\|_2, \sigma_p\sqrt{d}\}\big)$ and $\|\boldsymbol{\varphi}(\boldsymbol{x}_{n,l}\boldsymbol{W}_Q^{(t)}\boldsymbol{W}_K^{(t)\top}(\boldsymbol{X}_n)^\top)\|_2 = O(1)$. Taking a summation and get

$$\|\boldsymbol{W}_V^{(T_3)}\boldsymbol{w}_O\|_2 \leq \|\boldsymbol{W}_V^{(0)}\boldsymbol{w}_O\|_2 + \sum_{t=0}^{T_3-1} \|\boldsymbol{W}_V^{(t+1)}\boldsymbol{w}_O - \boldsymbol{W}_V^{(t)}\boldsymbol{w}_O\|_2$$

$$= \|\boldsymbol{W}_V^{(0)}\boldsymbol{w}_O\|_2 + O\Big(\frac{1}{\eta\epsilon\|\boldsymbol{\mu}\|_2^2\|\boldsymbol{w}_O\|_2^2}\Big) \cdot O\Big(\eta \cdot \max\{\|\boldsymbol{\mu}\|_2, \sigma_p\sqrt{d}\} \cdot \|\boldsymbol{w}_O\|_2^2\Big)$$

$$= O\Big(\sigma_V\|\boldsymbol{w}_O\|_2\sqrt{d}\Big) + O\Big(\frac{\max\{\|\boldsymbol{\mu}\|_2, \sigma_p\sqrt{d}\}}{\epsilon\|\boldsymbol{\mu}\|_2^2}\Big)$$

$$= O\Big(\sigma_V\|\boldsymbol{w}_O\|_2\sqrt{d} + \frac{\max\{\|\boldsymbol{\mu}\|_2, \sigma_p\sqrt{d}\}}{\epsilon\|\boldsymbol{\mu}\|_2^2}\Big).$$
(134)

Then $\boldsymbol{\xi}_i^\top \boldsymbol{W}_V^{(T_3)}\boldsymbol{w}_O$ is a Gaussian random variable with mean zero and standard deviation smaller than $O\Big(\sigma_V\|\boldsymbol{w}_O\|_2\sigma_p\sqrt{d} + \frac{\max\{\|\boldsymbol{\mu}\|_2, \sigma_p\sqrt{d}\}\cdot\sigma_p}{\epsilon\|\boldsymbol{\mu}\|_2^2}\Big)$. By Gaussian tail bound, for any $i \in [M]\backslash\{1\}$, with probability at least $1 - \delta/2N^2M^2$,

$$|\boldsymbol{\xi}_i^\top \boldsymbol{W}_V^{(T_3)}\boldsymbol{w}_O| \leq O\Big(\sigma_V\|\boldsymbol{w}_O\|_2\sigma_p\sqrt{d} + \frac{\max\{\|\boldsymbol{\mu}\|_2, \sigma_p\sqrt{d}\}\cdot\sigma_p}{\epsilon\|\boldsymbol{\mu}\|_2^2}\Big) \cdot \sqrt{2\log\big(4N^2M^2/\delta\big)}$$

$$\leq 1,$$

where the last inequality is by $\sigma_V \leq \widetilde{O}\big(\|\boldsymbol{w}_O\|_2^{-1}\cdot\min\{\|\boldsymbol{\mu}\|_2^{-1}, (\sigma_p\sqrt{d})^{-1}\}\cdot d_h^{-\frac{1}{4}}\big)$, $N\cdot\text{SNR}^2 = \Omega(1)$ and $d = \widetilde{\Omega}\big(\epsilon^{-2}N^2d_h\big)$ and $d_h = \widetilde{\Omega}\big(\max\{\text{SNR}^4, \text{SNR}^{-4}\}N^2\epsilon^{-2}\big)$. Applying a union bound completes the proof.

**Lemma D.10** (Population Loss). *Under the same conditions as Theorem 4.1, we have*

$$L_D\big(\theta(T_3)\big) = o(1)$$

*Proof of Lemma D.10.* Let event A to be the event that Lemma D.8 holds, and event B to be the event that Lemma D.9 holds. By a union bound, $\mathbb{P}(A \wedge B) \geq 1 - \delta/N^2M$. We decompose $L_D\big(\theta(T_3)\big)$ into the following form:

$$L_D\big(\theta(T_3)\big) = \mathbb{E}[\mathbb{1}(A \wedge B)\ell(yf(\boldsymbol{X}, \theta(T_3)))] + \mathbb{E}[\mathbb{1}((A \wedge B)^c)\ell(yf(\boldsymbol{X}, \theta(T_3)))] \quad (135)$$

Then we bound $\mathbb{E}[\mathbb{1}(A \wedge B)\ell(yf(\boldsymbol{X}, \theta(T_3)))]$ and $\mathbb{E}[\mathbb{1}((A \wedge B)^c)\ell(yf(\boldsymbol{X}, \theta(T_3)))]$ respectively.
**Bounding** $\mathbb{E}[\mathbb{1}(A \wedge B)\ell(yf(\boldsymbol{X}, \theta(T_3)))]$: When event A and event B hold, we have

$$y(f(\boldsymbol{X}, \theta(T_3))) = \frac{1}{M}\sum_{l=1}^{M} \boldsymbol{\varphi}(\boldsymbol{x}_l^\top \boldsymbol{W}_Q^{(T_3)}\boldsymbol{W}_K^{(T_3)\top}\boldsymbol{X}^\top)\boldsymbol{X}\boldsymbol{W}_V^{(T_3)}\boldsymbol{w}_O$$

$$\geq \frac{1}{2M^2}|\boldsymbol{\mu}_+^\top \boldsymbol{W}_V^{(T_3)}\boldsymbol{w}_O| - \big(1 - \frac{1}{2M^2}\big)\max\{|\boldsymbol{\xi}_i^\top \boldsymbol{W}_V^{(T_3)}\boldsymbol{w}_O|\}$$

$$\geq \frac{1}{2M^2}\log\Big(\frac{C_{20}}{\epsilon}\Big) - 1 \quad (136)$$

$$\geq \frac{\log\big(\frac{C_{20}}{\epsilon}\big)}{C_{21}} - 1,$$

where the first inequality is by Lemma D.8, the second inequality is by (112) and Lemma D.9, the last inequality is by $M = \Theta(1)$. Therefore,

$$
\begin{aligned}
\mathbb{E}[\mathbb{1}(A \wedge B)\ell(yf(\boldsymbol{X}, \theta(T_3)))] &\leq \ell(yf(\boldsymbol{X}, \theta(T_3))) \\
&\leq \log(1 + \exp(-yf(\boldsymbol{X}, \theta(T_3)))) \\
&\leq \exp(-yf(\boldsymbol{X}, \theta(T_3))) \\
&\leq \exp\left(1 - \frac{\log\left(\frac{C_{20}}{\epsilon}\right)}{C_{21}}\right) \\
&\leq 3\exp\left(\frac{\log\left(\frac{\epsilon}{C_{20}}\right)}{C_{21}}\right)
\end{aligned}
\tag{137}
$$

**Bounding** $\mathbb{E}[\mathbb{1}((A \wedge B)^c)\ell(yf(\boldsymbol{X}, \theta(T_3)))]$:

$$
\begin{aligned}
\mathbb{E}[\mathbb{1}((A \wedge B)^c)\ell(yf(\boldsymbol{X}, \theta(T_3)))] &\leq \sqrt{\mathbb{E}[\mathbb{1}((A \wedge B)^c)]} \cdot \sqrt{\mathbb{E}[\ell(yf(\boldsymbol{X}, \theta(T_3)))^2]} \\
&\leq \sqrt{\mathbb{P}((A \wedge B)^c)} \cdot \sqrt{\mathbb{E}[\exp(-2yf(\boldsymbol{X}, \theta(T_3)))]} \\
&\leq \frac{\delta^{\frac{1}{2}}}{N} \cdot \sqrt{\mathbb{E}[\exp(-2f(\boldsymbol{X}, \theta(T_3)))]},
\end{aligned}
\tag{138}
$$

where the first inequality is by Cauchy-Schwartz inequality, the second inequality is by $\log(1 + \exp(-z)) \leq \exp(-z)$.

Next we provide a bound for $\mathbb{E}[\exp(-2f(\boldsymbol{X}, \theta(T_3)))]$. Note that

$$
y(f(\boldsymbol{X}, \theta(T_3))) = \frac{1}{M} \sum_{l=1}^{M} \varphi(\boldsymbol{x}_l^\top \boldsymbol{W}_Q^{(T_3)} \boldsymbol{W}_K^{(T_3)\top} \boldsymbol{X}^\top) \boldsymbol{X} \boldsymbol{W}_V^{(T_3)} \boldsymbol{w}_O
$$

and $\boldsymbol{\mu}_+ \boldsymbol{W}_V^{T_3} \boldsymbol{w}_O \geq 0$, we further have

$$
y(f(\boldsymbol{X}, \theta(T_3))) \geq -\sum_{i=2}^{M} |\boldsymbol{\xi}_i \boldsymbol{W}_V^{(T_3)} \boldsymbol{w}_O|
$$

Therefore,

$$
\begin{aligned}
\mathbb{E}[\exp(-2f(\boldsymbol{X}, \theta(T_3)))] &\leq \mathbb{E}[\exp(2\sum_{i=2}^{M} |\boldsymbol{\xi}_i \boldsymbol{W}_V^{(T_3)} \boldsymbol{w}_O|)] \\
&\leq \mathbb{E}[\prod_{i=2}^{M} \exp(2 \cdot |\boldsymbol{\xi}_i \boldsymbol{W}_V^{(T_3)} \boldsymbol{w}_O|)] \\
&\leq \mathbb{E}[\prod_{i=2}^{M} \left(\exp(2\boldsymbol{\xi}_i \boldsymbol{W}_V^{(T_3)} \boldsymbol{w}_O) + \exp(-2\boldsymbol{\xi}_i \boldsymbol{W}_V^{(T_3)} \boldsymbol{w}_O)\right)] \\
&\leq \prod_{i=2}^{M} \mathbb{E}[\left(\exp(2\boldsymbol{\xi}_i \boldsymbol{W}_V^{(T_3)} \boldsymbol{w}_O) + \exp(-2\boldsymbol{\xi}_i \boldsymbol{W}_V^{(T_3)} \boldsymbol{w}_O)\right)] \\
&\leq 2^M \prod_{i=2}^{M} \mathbb{E}[\exp(2\boldsymbol{\xi}_i \boldsymbol{W}_V^{(T_3)} \boldsymbol{w}_O)]
\end{aligned}
\tag{139}
$$

where the third inequality is by $|x| \leq \max\{x, -x\}$, The fourth inequality is by the independence of $\boldsymbol{\xi}_i \boldsymbol{W}_V^{(T_3)} \boldsymbol{w}_O$.

We denote $\sigma = \|\boldsymbol{W}_V^{(T_3)}\boldsymbol{w}_O\|_2$, then $\boldsymbol{\xi}_i \boldsymbol{W}_V^{(T_3)}\boldsymbol{w}_O$ is a Gaussian distribution with mean zero and variance smaller than $C_p^2\sigma^2\sigma_p^2$. Therefore,

$$
\begin{aligned}
\mathbb{E}[\exp(2\boldsymbol{\xi}_i\boldsymbol{W}_V^{(T_3)}\boldsymbol{w}_O)] &\leq \exp(2C_p^2\sigma^2\sigma_p^2) \\
&= \exp\left(2C_p^2\sigma_p^2 \cdot O\left(\sigma_V^2\|\boldsymbol{w}_O\|_2^2 d + \frac{\max\{\|\boldsymbol{\mu}\|_2^2, \sigma_p^2 d\}}{\epsilon^2\|\boldsymbol{\mu}\|_2^4}\right)\right) \\
&= \exp\left(O\left(\sigma_V^2\|\boldsymbol{w}_O\|_2^2\sigma_p^2 d + \frac{\max\{\|\boldsymbol{\mu}\|_2^2, \sigma_p^2 d\}\cdot\sigma_p^2}{\epsilon^2\|\boldsymbol{\mu}\|_2^4}\right)\right) \\
&= \exp\left(o(1)\right) \\
&= 1 + o(1),
\end{aligned}
\tag{140}
$$

where the first equality is by (134), the third equality is by $\sigma_V \leq \widetilde{O}\big(\|\boldsymbol{w}_O\|_2^{-1} \cdot \min\{\|\boldsymbol{\mu}\|_2^{-1}, (\sigma_p\sqrt{d})^{-1}\} \cdot d_h^{-\frac{1}{4}}\big)$, $N \cdot \mathrm{SNR}^2 \geq \Omega(1)$ and $d = \widetilde{\Omega}\big(\epsilon^{-2}N^2 d_h\big)$. Then we have

$$
\begin{aligned}
\mathbb{E}[\exp(-2f(\boldsymbol{X}, \theta(T_3)))] &\leq 2^M \prod_{i=2}^{M} \mathbb{E}[\exp(2\boldsymbol{\xi}_i\boldsymbol{W}_V^{(T_3)}\boldsymbol{w}_O)] \\
&\leq 2^M \cdot (1 + o(1))^M \\
&= O(1).
\end{aligned}
\tag{141}
$$

We further get

$$
\mathbb{E}[\mathbb{1}((A \wedge B)^c)\ell(yf(\boldsymbol{X}, \theta(T_3)))] \leq \frac{\delta^{\frac{1}{2}}}{N} \cdot \sqrt{\mathbb{E}[\exp(-2f(\boldsymbol{X}, \theta(T_3)))]} \leq \frac{C_{24}\delta^{\frac{1}{2}}}{N} \tag{142}
$$

Plugging (137) and (142) into (135), we have

$$
L_D\big(\theta(T_3)\big) \leq 3\exp\left(\frac{\log\left(\frac{\epsilon}{C_{20}}\right)}{C_{21}}\right) + \frac{C_{24}\delta^{\frac{1}{2}}}{N} = o(1).
$$

## E  Harmful Overfitting

In this section, we consider harmful overfitting case under the condition that $N^{-1} \cdot \mathrm{SNR}^{-2} \geq \Omega(1)$. The proofs in this section are based on the results in Section C, which hold with high probability.

### E.1  Stage I

In Stage I, $V_\pm^{(t)}$, $V_{n,i}^{(t)}$ begin to pull apart until $|V_\pm^{(t)}|$ is sufficiently larger than $|V_{n,i}^{(t)}|$. At the same time, the inner products of $\boldsymbol{q}$ and $\boldsymbol{k}$ maintain their magnitude.

**Lemma E.1** (Upper bound of V)**.** *Let* $T_0 = O(\frac{N}{\eta d_h^{\frac{1}{4}}\sigma_p^2 d\|\boldsymbol{w}_O\|_2^2})$*. Then under the same conditions as Theorem 4.2, we have*

$$
|V_+^{(t)}|, |V_-^{(t)}|, |V_{n,i}^{(t)}| = O(d_h^{-\frac{1}{4}})
$$

*for* $t \in [0, T_0]$.

*Proof of Lemma E.1.* By Lemma B.2, we have

$$
\begin{aligned}
&|\gamma_{V,+}^{(t+1)} - \gamma_{V,+}^{(t)}| \\
&\leq -\frac{\eta\|\boldsymbol{\mu}\|_2^2}{NM} \sum_{n\in S_+} \ell_n'^{(t)} \Big( \frac{\exp(\langle \boldsymbol{q}_+^{(t)}, \boldsymbol{k}_+^{(t)}\rangle)}{\exp(\langle \boldsymbol{q}_+^{(t)}, \boldsymbol{k}_+^{(t)}\rangle) + \sum\limits_{k=2}^{M} \exp(\langle \boldsymbol{q}_+^{(t)}, \boldsymbol{k}_{n,k}^{(t)}\rangle)} \\
&\quad + \sum_{j=2}^{M} \frac{\exp(\langle \boldsymbol{q}_{n,j}^{(t)}, \boldsymbol{k}_+^{(t)}\rangle)}{\exp(\langle \boldsymbol{q}_{n,j}^{(t)}, \boldsymbol{k}_+^{(t)}\rangle) + \sum\limits_{k=2}^{M} \exp(\langle \boldsymbol{q}_{n,j}^{(t)}, \boldsymbol{k}_{n,k}^{(t)}\rangle)} \Big) \\
&\leq \frac{\eta\|\boldsymbol{\mu}\|_2^2}{NM} \cdot \frac{3N}{4}\big(1 + (M-1)\big) \\
&\leq \frac{3\eta\|\boldsymbol{\mu}\|_2^2}{4},
\end{aligned}
$$

where the second inequality is by Lemma C.1 and $-\ell_n'^{(t)} \leq 1$. Similarly, we have

$$
|\gamma_{V,-}^{(t+1)} - \gamma_{V,-}^{(t)}| \leq \frac{3\eta\|\boldsymbol{\mu}\|_2^2}{4}.
$$

By Definition B.1, we have

$$
\begin{aligned}
|V_+^{(t)}| &= \Big| V_+^{(0)} + \sum_{s=0}^{t-1} (\gamma_{V,+}^{(s+1)} - \gamma_{V,+}^{(s)})\|\boldsymbol{w}_O\|_2^2 \Big| \\
&\leq |V_+^{(0)}| + \sum_{s=0}^{T_0-1} |\gamma_{V,+}^{(s+1)} - \gamma_{V,+}^{(s)}| \cdot \|\boldsymbol{w}_O\|_2^2 \\
&\leq d_h^{-\frac{1}{4}} + \frac{3\eta\|\boldsymbol{\mu}\|_2^2}{4} \cdot \|\boldsymbol{w}_O\|_2^2 \cdot O\Big(\frac{N}{\eta d_h^{\frac{1}{4}}\sigma_p^2 d\|\boldsymbol{w}_O\|_2^2}\Big) \\
&= d_h^{-\frac{1}{4}} + O\Big(\frac{N\|\boldsymbol{\mu}\|_2^2}{d_h^{\frac{1}{4}}\sigma_p^2 d}\Big) \\
&= O(d_h^{-\frac{1}{4}}),
\end{aligned}
$$

where the first inequality is by triangle inequality and $t \leq T_0$, the second inequality is by Lemma C.2, the last equality is by $N^{-1} \cdot \mathrm{SNR}^{-2} = \Omega(1)$. Similarly, we have $|V_-^{(t)}| = O(d_h^{-\frac{1}{4}})$.

By Lemma B.2, we have

$$|\rho_{V,n,i}^{(t+1)} - \rho_{V,n,i}^{(t)}|$$

$$\leq \Big| - \frac{\eta}{NM} \sum_{n' \in S_+} \ell_{n'}^{\prime(t)} \Big( \sum_{i=2}^{M} (\langle \boldsymbol{\xi}_{n,i}, \boldsymbol{\xi}_{n',i'} \rangle \frac{\exp(\langle \boldsymbol{q}_+^{(t)}, \boldsymbol{k}_{n',i'}^{(t)} \rangle)}{\exp(\langle \boldsymbol{q}_+^{(t)}, \boldsymbol{k}_+^{(t)} \rangle) + \sum\limits_{k=2}^{M} \exp(\langle \boldsymbol{q}_+^{(t)}, \boldsymbol{k}_{n',k}^{(t)} \rangle)}$$

$$+ \sum_{j=2}^{M} \langle \boldsymbol{\xi}_{n,i}, \boldsymbol{\xi}_{n',i'} \rangle \frac{\exp(\langle \boldsymbol{q}_{n',j}^{(t)}, \boldsymbol{k}_{n',i'}^{(t)} \rangle)}{\exp(\langle \boldsymbol{q}_{n',j}^{(t)}, \boldsymbol{k}_+^{(t)} \rangle) + \sum\limits_{k=2}^{M} \exp(\langle \boldsymbol{q}_{n',j}^{(t)}, \boldsymbol{k}_{n',k}^{(t)} \rangle)}))$$

$$+ \frac{\eta}{NM} \sum_{n' \in S_-} \ell_{n'}^{\prime(t)} \Big( \sum_{i=2}^{M} (\langle \boldsymbol{\xi}_{n,i}, \boldsymbol{\xi}_{n',i'} \rangle \frac{\exp(\langle \boldsymbol{q}_-^{(t)}, \boldsymbol{k}_{n',i'}^{(t)} \rangle)}{\exp(\langle \boldsymbol{q}_-^{(t)}, \boldsymbol{k}_-^{(t)} \rangle) + \sum\limits_{k=2}^{M} \exp(\langle \boldsymbol{q}_-^{(t)}, \boldsymbol{k}_{n',k}^{(t)} \rangle)} \qquad (143)$$

$$+ \sum_{j=2}^{M} \langle \boldsymbol{\xi}_{n,i}, \boldsymbol{\xi}_{n',i'} \rangle \frac{\exp(\langle \boldsymbol{q}_{n',j}^{(t)}, \boldsymbol{k}_{n',i'}^{(t)} \rangle)}{\exp(\langle \boldsymbol{q}_{n',j}^{(t)}, \boldsymbol{k}_-^{(t)} \rangle) + \sum\limits_{k=2}^{M} \exp(\langle \boldsymbol{q}_{n',j}^{(t)}, \boldsymbol{k}_{n',k}^{(t)} \rangle)})) \Big|$$

$$\leq \frac{3\eta \tilde{\sigma}_p^2 d}{2NM} \cdot M + \frac{\eta}{NM} \cdot MN \cdot 2\tilde{\sigma}_p^2 \cdot \sqrt{d \log(4N^2 M^2/\delta)}$$

$$\leq \frac{2\eta \tilde{\sigma}_p^2 d}{N}$$

$$= \frac{2\eta C_p^2 \sigma_p^2 d}{N}$$

where the second inequality is by Lemma C.4 and $-\ell_n^{\prime(t)} \leq 1$, the third inequality is by $d = \widetilde{\Omega}\Big(\epsilon^{-2} N^2 d_h\Big)$, the last inequality is by $N \cdot \mathrm{SNR}^2 = \Omega(1)$. Then by Definition B.1, we have

$$|V_{n,i}^{(t)}| = |V_{n,i}^{(0)} + \sum_{s=0}^{t-1} (\gamma_{V,n,i}^{(s+1)} - \gamma_{V,n,i}^{(s)}) \|\boldsymbol{w}_O\|_2^2|$$

$$\leq |V_{n,i}^{(0)}| + \sum_{s=0}^{t-1} |\gamma_{V,n,i}^{(s+1)} - \gamma_{V,n,i}^{(s)}| \cdot \|\boldsymbol{w}_O\|_2^2$$

$$\leq d_h^{-\frac{1}{4}} + \frac{2\eta C_p^2 \sigma_p^2 d}{N} \cdot \|\boldsymbol{w}_O\|_2^2 \cdot O(\frac{N}{\eta d_h^{\frac{1}{4}} \sigma_p^2 d \|\boldsymbol{w}_O\|_2^2})$$

$$= O(d^{-\frac{1}{4}}),$$

where the first inequality is by triangle inequality, the second inequality is by Lemma C.2, which completes the proof.

**Lemma E.2** (Inner Products Hold Magnitude). *Let* $T_0 = O(\frac{N}{\eta d_h^{\frac{1}{4}} \sigma_p^2 d \|\boldsymbol{w}_O\|_2^2})$. *Then under the same conditions as Theorem 4.2, we have*

$$|\langle \boldsymbol{q}_\pm^{(t)}, \boldsymbol{k}_\pm^{(t)} \rangle|, |\langle \boldsymbol{q}_{n,i}^{(t)}, \boldsymbol{k}_\pm^{(t)} \rangle|, |\langle \boldsymbol{q}_\pm^{(t)}, \boldsymbol{k}_{n,j}^{(t)} \rangle|, |\langle \boldsymbol{q}_{n,i}^{(t)}, \boldsymbol{k}_{n',j}^{(t)} \rangle|$$

$$= O\Big( \max\{\|\boldsymbol{\mu}\|_2^2, \sigma_p^2 d\} \cdot \sigma_h^2 \cdot \sqrt{d_h \log(6N^2 M^2/\delta)} \Big),$$

$$|\langle \boldsymbol{q}_\pm^{(t)}, \boldsymbol{q}_\mp^{(t)} \rangle|, |\langle \boldsymbol{q}_{n,i}^{(t)}, \boldsymbol{q}_\pm^{(t)} \rangle|, |\langle \boldsymbol{q}_{n,i}^{(t)}, \boldsymbol{q}_{n',j}^{(t)} \rangle|$$

$$= O\Big( \max\{\|\boldsymbol{\mu}\|_2^2, \sigma_p^2 d\} \cdot \sigma_h^2 \cdot \sqrt{d_h \log(6N^2 M^2/\delta)} \Big),$$

$$|\langle \boldsymbol{k}_\pm^{(t)}, \boldsymbol{k}_\mp^{(t)} \rangle|, |\langle \boldsymbol{k}_{n,i}^{(t)}, \boldsymbol{k}_\pm^{(t)} \rangle|, |\langle \boldsymbol{k}_{n,i}^{(t)}, \boldsymbol{k}_{n',j}^{(t)} \rangle|$$

$$= O\left(\max\{\|\boldsymbol{\mu}\|_2^2, \sigma_p^2 d\} \cdot \sigma_h^2 \cdot \sqrt{d_h \log(6N^2 M^2/\delta)}\right),$$

$$\|\boldsymbol{q}_\pm^{(t)}\|_2^2, \|\boldsymbol{k}_\pm^{(t)}\|_2^2 = \Theta(\|\boldsymbol{\mu}\|_2^2 \sigma_h^2 d_h),$$

$$\|\boldsymbol{q}_{n,i}^{(t)}\|_2^2, \|\boldsymbol{k}_{n,i}^{(t)}\|_2^2 = \Theta(\sigma_p^2 \sigma_h^2 d d_h)$$

for $i, j \in [M]\backslash\{1\}$, $n, n' \in [N]$ and $t \in [0, T_0]$.

The proof for Lemma E.2 is similar to D.4.

**Lemma E.3** (V's Beginning of Learning Signals). *There exist $T_1 = \dfrac{4M(3M+1)N}{\eta d_h^{\frac{1}{4}}(C_p^2 - 24M)\sigma_p^2 d\|\boldsymbol{w}_O\|_2^2}$ such that the second element of vector $\boldsymbol{X}_n \boldsymbol{W}_V^{(T_1)} \boldsymbol{w}_O$ dominate its other elements, which means $V_{n,2}^{(T_1)} \geq 3M \cdot \max\{|V_+^{(T_1)}|, |V_{n,i}^{(T_1)}|\}$ for all $n \in S_+$, $i \in [M]\backslash\{1, 2\}$ and $V_{n,2}^{(T_1)} \leq -3M \cdot \max\{|V_-^{(T_1)}|, |V_{n,i}^{(T_1)}|\}$ for all $n \in S_-$, $i \in [M]\backslash\{1, 2\}$.*

*Proof of Lemma E.3.* According to Condition 4.1 where $C_p^2 = 25M$, we have Then we have

$$T_1 = \frac{4M(3M+1)N}{\eta d_h^{\frac{1}{4}}(25M - 24M)\sigma_p^2 d\|\boldsymbol{w}_O\|_2^2} = \frac{4(3M+1)N}{\eta d_h^{\frac{1}{4}}\sigma_p^2 d\|\boldsymbol{w}_O\|_2^2} = O\left(\frac{N}{\eta d_h^{\frac{1}{4}}\sigma_p^2 d\|\boldsymbol{w}_O\|_2^2}\right),$$

which satisfies the time condition in Lemma E.1 and Lemma E.2. Then by Lemma D.1 and Lemma D.2 we have

$$-\ell_n'^{(t)} = \frac{1}{2} \pm o(1),$$

$$\frac{1}{M} - o(1) \leq softmax(\langle \boldsymbol{q}_\pm^{(t)}, \boldsymbol{k}_\pm^{(t)}\rangle) \leq \frac{1}{M} + o(1),$$

$$\frac{1}{M} - o(1) \leq softmax(\langle \boldsymbol{q}_{n,i}^{(t)}, \boldsymbol{k}_\pm^{(t)}\rangle) \leq \frac{1}{M} + o(1),$$

$$\frac{1}{M} - o(1) \leq softmax(\langle \boldsymbol{q}_\pm^{(t)}, \boldsymbol{k}_{n,j}^{(t)}\rangle) \leq \frac{1}{M} + o(1),$$

$$\frac{1}{M} - o(1) \leq softmax(\langle \boldsymbol{q}_{n,i}^{(t)}, \boldsymbol{k}_{n,j}^{(t)}\rangle) \leq \frac{1}{M} + o(1)$$

for $i, j \in [M]\backslash\{1\}$, $n \in [N]$ and $t \in [0, T_1]$. Plugging them in the update rule for $\gamma_{V,+}^{(t)}$ showed in Lemma B.2 and we have

$$
\begin{aligned}
&|\gamma_{V,+}^{(t+1)} - \gamma_{V,+}^{(t)}| \\
&= \left| -\frac{\eta\|\boldsymbol{\mu}\|_2^2}{NM} \sum_{n \in S_+} \ell_n'^{(t)} \left( \frac{\exp(\langle \boldsymbol{q}_+^{(t)}, \boldsymbol{k}_+^{(t)}\rangle)}{\exp(\langle \boldsymbol{q}_+^{(t)}, \boldsymbol{k}_+^{(t)}\rangle) + \sum_{k=2}^{M}\exp(\langle \boldsymbol{q}_+^{(t)}, \boldsymbol{k}_{n,k}^{(t)}\rangle)} \right. \right. \\
&\quad \left. \left. + \sum_{j=2}^{M} \frac{\exp(\langle \boldsymbol{q}_{n,j}^{(t)}, \boldsymbol{k}_+^{(t)}\rangle)}{\exp(\langle \boldsymbol{q}_{n,j}^{(t)}, \boldsymbol{k}_+^{(t)}\rangle) + \sum_{k=2}^{M}\exp(\langle \boldsymbol{q}_{n,j}^{(t)}, \boldsymbol{k}_{n,k}^{(t)}\rangle)} \right) \right| \\
&\leq \frac{\eta\|\boldsymbol{\mu}\|_2^2}{NM} \cdot \frac{3N}{4} \cdot \left(\frac{1}{2} \pm o(1)\right) \cdot M\left(\frac{1}{M} \pm o(1)\right) \\
&\leq \frac{\eta\|\boldsymbol{\mu}\|_2^2}{2M} \\
&\leq \frac{2\eta\sigma_p^2 d}{NM}
\end{aligned}
\tag{144}
$$

where the last inequality is by $N^{-1} \cdot \text{SNR}^{-2} = \Omega(1)$. Then by Definition B.1 and taking a summation, we have

$$
\begin{aligned}
|V_+^{(T_1)}| &\leq |V_+^{(0)}| + T_1 \frac{2\eta\sigma_p^2 d}{NM}\|\boldsymbol{w}_O\|_2^2 \\
&= d_h^{-\frac{1}{4}} + \frac{4M(3M+1)N}{\eta d_h^{\frac{1}{4}}(C_p^2 - 24M)\sigma_p^2 d\|\boldsymbol{w}_O\|_2^2} \cdot \frac{2\eta\sigma_p^2 d}{NM}\|\boldsymbol{w}_O\|_2^2 \\
&= d_h^{-\frac{1}{4}} + \frac{8(3M+1)}{d_h^{\frac{1}{4}}(C_p^2 - 24M)}.
\end{aligned}
\tag{145}
$$

Similarly, we have

$$
|V_-^{(T_1)}| \leq d_h^{-\frac{1}{4}} + \frac{8(3M+1)}{d_h^{\frac{1}{4}}(C_p^2 - 24M)}.
\tag{146}
$$

For $n \in S_+$, by Lemma B.2, we have

$$
\begin{aligned}
&\rho_{V,n,2}^{(t+1)} - \rho_{V,n,2}^{(t)} \\
&= -\frac{\eta}{NM} \sum_{n' \in S_+} \ell_{n'}^{\prime(t)} \Big( \sum_{i'=2}^{M} \big(\langle \boldsymbol{\xi}_{n,2}, \boldsymbol{\xi}_{n',i'} \rangle \frac{\exp(\langle \boldsymbol{q}_+^{(t)}, \boldsymbol{k}_{n',i'}^{(t)} \rangle)}{\exp(\langle \boldsymbol{q}_+^{(t)}, \boldsymbol{k}_+^{(t)} \rangle) + \sum\limits_{k=2}^{M} \exp(\langle \boldsymbol{q}_+^{(t)}, \boldsymbol{k}_{n',k}^{(t)} \rangle)} \\
&\quad + \sum_{j=2}^{M} \langle \boldsymbol{\xi}_{n,2}, \boldsymbol{\xi}_{n',i'} \rangle \frac{\exp(\langle \boldsymbol{q}_{n',j}^{(t)}, \boldsymbol{k}_{n',i'}^{(t)} \rangle)}{\exp(\langle \boldsymbol{q}_{n',j}^{(t)}, \boldsymbol{k}_+^{(t)} \rangle) + \sum\limits_{k=2}^{M} \exp(\langle \boldsymbol{q}_{n',j}^{(t)}, \boldsymbol{k}_{n',k}^{(t)} \rangle)} \big) \Big) \\
&\quad + \frac{\eta}{NM} \sum_{n' \in S_-} \ell_{n'}^{\prime(t)} \Big( \sum_{i'=2}^{M} \big(\langle \boldsymbol{\xi}_{n,2}, \boldsymbol{\xi}_{n',i'} \rangle \frac{\exp(\langle \boldsymbol{q}_-^{(t)}, \boldsymbol{k}_{n',i'}^{(t)} \rangle)}{\exp(\langle \boldsymbol{q}_-^{(t)}, \boldsymbol{k}_-^{(t)} \rangle) + \sum\limits_{k=2}^{M} \exp(\langle \boldsymbol{q}_-^{(t)}, \boldsymbol{k}_{n',k}^{(t)} \rangle)} \\
&\quad + \sum_{j=2}^{M} \langle \boldsymbol{\xi}_{n,2}, \boldsymbol{\xi}_{n',i'} \rangle \frac{\exp(\langle \boldsymbol{q}_{n',j}^{(t)}, \boldsymbol{k}_{n',i'}^{(t)} \rangle)}{\exp(\langle \boldsymbol{q}_{n',j}^{(t)}, \boldsymbol{k}_-^{(t)} \rangle) + \sum\limits_{k=2}^{M} \exp(\langle \boldsymbol{q}_{n',j}^{(t)}, \boldsymbol{k}_{n',k}^{(t)} \rangle)} \big) \Big) \\
&\geq \frac{\eta\tilde{\sigma}_p^2 d}{2NM} \cdot M\Big(\frac{1}{M} - o(1)\Big) - \frac{\eta}{NM} \cdot MN \cdot 2\tilde{\sigma}_p^2 \cdot \sqrt{d\log(4N^2M^2/\delta)} \\
&\geq \frac{\eta\tilde{\sigma}_p^2 d}{4NM} \\
&\geq \frac{\eta C_p^2 \sigma_p^2 d}{4NM}
\end{aligned}
\tag{147}
$$

where the second inequality is by Lemma C.4 and $-\ell_n^{\prime(t)} \leq 1$, the third inequality is by $d = \widetilde{\Omega}\big(\epsilon^{-2}N^2 d_h\big)$. Then by Definition B.1, we have

$$
\begin{aligned}
V_{n,2}^{(T_1)} &= V_{n,2}^{(0)} + \sum_{s=0}^{T_1-1} (\rho_{V,n,2}^{(s+1)} - \rho_{V,n,2}^{(s)})\|\boldsymbol{w}_O\|_2^2 \\
&\geq -d_h^{-\frac{1}{4}} + \frac{4M(3M+1)N}{\eta d_h^{\frac{1}{4}}(C_p^2 - 24M)\sigma_p^2 d\|\boldsymbol{w}_O\|_2^2} \cdot \frac{\eta C_p^2 \sigma_p^2 d}{4NM}\|\boldsymbol{w}_O\|_2^2 \\
&\geq -d_h^{-\frac{1}{4}} + \frac{C_p^2(3M+1)}{d_h^{\frac{1}{4}}(C_p^2 - 24M)}
\end{aligned}
\tag{148}
$$

for $n \in S_+$. Similarly, for $n \in S_-$, we have

$$
V_{n,2}^{(T_1)} \leq d_h^{-\frac{1}{4}} - \frac{C_p^2(3M+1)}{d_h^{\frac{1}{4}}(C_p^2 - 24M)}.
\tag{149}
$$

By Lemma B.2, we have

$$
|\rho_{V,n,i}^{(t+1)} - \rho_{V,n,i}^{(t)}|
$$

$$
\leq \Big| -\frac{\eta}{NM}\sum_{n'\in S_+}\ell_{n'}^{\prime(t)}\Big(\sum_{i=2}^{M}(\langle\boldsymbol{\xi}_{n,i},\boldsymbol{\xi}_{n',i'}\rangle\frac{\exp(\langle\boldsymbol{q}_+^{(t)},\boldsymbol{k}_{n',i'}^{(t)}\rangle)}{\exp(\langle\boldsymbol{q}_+^{(t)},\boldsymbol{k}_+^{(t)}\rangle)+\sum\limits_{k=2}^{M}\exp(\langle\boldsymbol{q}_+^{(t)},\boldsymbol{k}_{n',k}^{(t)}\rangle)}
$$

$$
+\sum_{j=2}^{M}\langle\boldsymbol{\xi}_{n,i},\boldsymbol{\xi}_{n',i'}\rangle\frac{\exp(\langle\boldsymbol{q}_{n',j}^{(t)},\boldsymbol{k}_{n',i'}^{(t)}\rangle)}{\exp(\langle\boldsymbol{q}_{n',j}^{(t)},\boldsymbol{k}_+^{(t)}\rangle)+\sum\limits_{k=2}^{M}\exp(\langle\boldsymbol{q}_{n',j}^{(t)},\boldsymbol{k}_{n',k}^{(t)}\rangle)}))
$$

$$
+\frac{\eta}{NM}\sum_{n'\in S_-}\ell_{n'}^{\prime(t)}\Big(\sum_{i=2}^{M}(\langle\boldsymbol{\xi}_{n,i},\boldsymbol{\xi}_{n',i'}\rangle\frac{\exp(\langle\boldsymbol{q}_-^{(t)},\boldsymbol{k}_{n',i'}^{(t)}\rangle)}{\exp(\langle\boldsymbol{q}_-^{(t)},\boldsymbol{k}_-^{(t)}\rangle)+\sum\limits_{k=2}^{M}\exp(\langle\boldsymbol{q}_-^{(t)},\boldsymbol{k}_{n',k}^{(t)}\rangle)} \qquad (150)
$$

$$
+\sum_{j=2}^{M}\langle\boldsymbol{\xi}_{n,i},\boldsymbol{\xi}_{n',i'}\rangle\frac{\exp(\langle\boldsymbol{q}_{n',j}^{(t)},\boldsymbol{k}_{n',i'}^{(t)}\rangle)}{\exp(\langle\boldsymbol{q}_{n',j}^{(t)},\boldsymbol{k}_-^{(t)}\rangle)+\sum\limits_{k=2}^{M}\exp(\langle\boldsymbol{q}_{n',j}^{(t)},\boldsymbol{k}_{n',k}^{(t)}\rangle)}))\Big|
$$

$$
\leq \frac{3\eta\sigma_p^2 d}{2NM}\cdot M(\frac{1}{M}+o(1))+\frac{\eta}{NM}\cdot MN\cdot 2\tilde{\sigma}_p^2\cdot\sqrt{d\log(4N^2M^2/\delta)}
$$

$$
\leq \frac{2\eta\sigma_p^2 d}{NM}
$$

for $i\in[M]\backslash\{1,2\}$, where the second inequality is by Lemma C.4 and $-\ell_n^{\prime(t)}\leq 1$, the last inequality is by $d=\widetilde{\Omega}\Big(\epsilon^{-2}N^2 d_h\Big)$. Then by Definition B.1, we have

$$
|V_{n,i}^{(t)}| = \Big|V_{n,i}^{(0)}+\sum_{s=0}^{t-1}(\gamma_{V,n,i}^{(s+1)}-\gamma_{V,n,i}^{(s)})\|\boldsymbol{w}_O\|_2^2\Big|
$$

$$
\leq |V_{n,i}^{(0)}|+\sum_{s=0}^{t-1}|\gamma_{V,n,i}^{(s+1)}-\gamma_{V,n,i}^{(s)}|\cdot\|\boldsymbol{w}_O\|_2^2 \qquad (151)
$$

$$
\leq d_h^{-\frac{1}{4}}+\frac{4M(3M+1)N}{\eta d_h^{\frac{1}{4}}(C_p^2-24M)\sigma_p^2 d\|\boldsymbol{w}_O\|_2^2}\cdot\frac{2\eta\sigma_p^2 d}{NM}\cdot\|\boldsymbol{w}_O\|_2^2
$$

$$
\leq d_h^{-\frac{1}{4}}+\frac{8(3M+1)}{d_h^{\frac{1}{4}}(C_p^2-24M)}
$$

for $i\in[M]\backslash\{1,2\}$, where the first inequality is by triangle inequality, the second inequality is by Lemma C.2.

According to (145), (146), (148), (149) and (151), it is easy to verify that $V_{n,2}^{(T_1)}\geq 3M\cdot\max\{|V_+^{(T_1)}|,|V_{n,i}^{(T_1)}|\}$ for all $n\in S_+$, $i\in[M]\backslash\{1,2\}$ and $V_{n,2}^{(T_1)}\leq -3M\cdot\max\{|V_-^{(T_1)}|,|V_{n,i}^{(T_1)}|\}$ for all $n\in S_-$, $i\in[M]\backslash\{1,2\}$, which completes the proof.

### E.2 Stage II

In stage II, $\langle\boldsymbol{q}_+,\boldsymbol{k}_+\rangle$, $\langle\boldsymbol{q}_{n,i},\boldsymbol{k}_+\rangle$ grows while $\langle\boldsymbol{q}_+,\boldsymbol{k}_{n,j}\rangle$, $\langle\boldsymbol{q}_{n,i},\boldsymbol{k}_{n,j}\rangle$ decreases, resulting in attention focusing more and more on the signal and less on the noise. By the results of stage I, we have the following condition at the beginning of stage II

$$
V_{n,2}^{(T_1)}\geq 3M\cdot\max\{|V_+^{(T_1)}|,|V_{n,i}^{(T_1)}|\}
$$

for all $n\in S_+$, $i\in[M]\backslash\{1,2\}$.

$$
V_{n,2}^{(T_1)}\leq -3M\cdot\max\{|V_-^{(T_1)}|,|V_{n,i}^{(T_1)}|\}
$$

for all $n \in S_-, i \in [M]\backslash\{1,2\}$.

$$|V_+^{(T_1)}|, |V_-^{(T_1)}|, |V_{n,i}^{(T_1)}| = O(d_h^{-\frac{1}{4}}),$$

$$|\langle \boldsymbol{q}_\pm^{(T_1)}, \boldsymbol{k}_\pm^{(T_1)}\rangle|, |\langle \boldsymbol{q}_{n,i}^{(T_1)}, \boldsymbol{k}_\pm^{(T_1)}\rangle|, |\langle \boldsymbol{q}_\pm^{(T_1)}, \boldsymbol{k}_{n,j}^{(T_1)}\rangle|, |\langle \boldsymbol{q}_{n,i}^{(T_1)}, \boldsymbol{k}_{n',j}^{(T_1)}\rangle|$$
$$= O\Big( \max\{\|\boldsymbol{\mu}\|_2^2, \sigma_p^2 d\} \cdot \sigma_h^2 \cdot \sqrt{d_h \log(6N^2 M^2/\delta)}\Big),$$

$$|\langle \boldsymbol{q}_\pm^{(T_1)}, \boldsymbol{q}_\mp^{(T_1)}\rangle|, |\langle \boldsymbol{q}_{n,i}^{(T_1)}, \boldsymbol{q}_\pm^{(T_1)}\rangle|, |\langle \boldsymbol{q}_{n,i}^{(T_1)}, \boldsymbol{q}_{n',j}^{(T_1)}\rangle|$$
$$= O\Big( \max\{\|\boldsymbol{\mu}\|_2^2, \sigma_p^2 d\} \cdot \sigma_h^2 \cdot \sqrt{d_h \log(6N^2 M^2/\delta)}\Big),$$

$$|\langle \boldsymbol{k}_\pm^{(T_1)}, \boldsymbol{k}_\mp^{(T_1)}\rangle|, |\langle \boldsymbol{k}_{n,i}^{(T_1)}, \boldsymbol{k}_\pm^{(T_1)}\rangle|, |\langle \boldsymbol{k}_{n,i}^{(T_1)}, \boldsymbol{k}_{n',j}^{(T_1)}\rangle|$$
$$= O\Big( \max\{\|\boldsymbol{\mu}\|_2^2, \sigma_p^2 d\} \cdot \sigma_h^2 \cdot \sqrt{d_h \log(6N^2 M^2/\delta)}\Big),$$

$$\|\boldsymbol{q}_\pm^{(T_1)}\|_2^2, \|\boldsymbol{k}_\pm^{(T_1)}\|_2^2 = \Theta(\|\boldsymbol{\mu}\|_2^2 \sigma_h^2 d_h),$$
$$\|\boldsymbol{q}_{n,i}^{(T_1)}\|_2^2, \|\boldsymbol{k}_{n,i}^{(T_1)}\|_2^2 = \Theta(\sigma_p^2 \sigma_h^2 d d_h)$$

for $i, j \in [M]\backslash\{1\}, n, n' \in [N]$.

Some of the proofs at this stage are based on the above conditions.

**Notations.** To better characterize the gap between different inner products, we define the following notations:

- denote $\Psi_+^{(t)} = \langle \boldsymbol{q}_+^{(t)}, \boldsymbol{k}_{n,2}^{(t)}\rangle - \langle \boldsymbol{q}_+^{(t)}, \boldsymbol{k}_+^{(t)}\rangle, \quad n \in S_+.$
- denote $\Psi_{n,+,j}^{(t)} = \langle \boldsymbol{q}_+^{(t)}, \boldsymbol{k}_{n,2}^{(t)}\rangle - \langle \boldsymbol{q}_+^{(t)}, \boldsymbol{k}_{n,j}^{(t)}\rangle, \quad n \in S_+, j \in [M]\backslash\{1,2\}.$
- denote $\Psi_-^{(t)} = \langle \boldsymbol{q}_-^{(t)}, \boldsymbol{k}_{n,2}^{(t)}\rangle - \langle \boldsymbol{q}_-^{(t)}, \boldsymbol{k}_-^{(t)}\rangle, \quad n \in S_-.$
- denote $\Psi_{n,-,j}^{(t)} = \langle \boldsymbol{q}_-^{(t)}, \boldsymbol{k}_{n,2}^{(t)}\rangle - \langle \boldsymbol{q}_-^{(t)}, \boldsymbol{k}_{n,j}^{(t)}\rangle, \quad n \in S_-, j \in [M]\backslash\{1,2\}.$
- denote $\Psi_{n,i,+}^{(t)} = \langle \boldsymbol{q}_{n,i}^{(t)}, \boldsymbol{k}_{n,2}^{(t)}\rangle - \langle \boldsymbol{q}_{n,i}^{(t)}, \boldsymbol{k}_+^{(t)}\rangle, \quad n \in S_+, i \in [M]\backslash\{1\}.$
- denote $\Psi_{n,i,-}^{(t)} = \langle \boldsymbol{q}_{n,i}^{(t)}, \boldsymbol{k}_{n,2}^{(t)}\rangle - \langle \boldsymbol{q}_{n,i}^{(t)}, \boldsymbol{k}_-^{(t)}\rangle, \quad n \in S_-, i \in [M]\backslash\{1\}.$
- denote $\Psi_{n,i,j}^{(t)} = \langle \boldsymbol{q}_{n,i}^{(t)}, \boldsymbol{k}_{n,2}^{(t)}\rangle - \langle \boldsymbol{q}_{n,i}^{(t)}, \boldsymbol{k}_{n,j}^{(t)}\rangle, i \in [M]\backslash\{1\}, j \in [M]\backslash\{1,2\}.$

**Lemma E.4** (Upper bound of V). *Let $T_0 = O\Big(\frac{N}{\eta \sigma_p^2 d\|\boldsymbol{w}_O\|_2^2 \log(6N^2 M^2/\delta)}\Big)$. Then under the same conditions as Theorem 4.2, we have*

$$|V_+^{(t)}|, |V_-^{(t)}|, |V_{n,i}^{(t)}| = o(1)$$

*for $t \in [0, T_0]$.*

The proof of Lemma E.4 is similar to that of Lemma E.1, except that the time $T_0$ is changed.

Let $T_2 = \Theta\Big(\frac{N}{\eta \sigma_p^2 d\|\boldsymbol{w}_O\|_2^2 \log(6N^2 M^2/\delta)}\Big)$, then by Lemma D.6 and Lemma D.1 we have $\frac{1}{2} - o(1) \leq -\ell_n'^{(t)} \leq \frac{1}{2} + o(1)$ for $n \in [N], t \in [T_1, T_2]$, which can simplify the calculations of $\alpha$ and $\beta$ defined in Definition B.3 by replacing $-\ell_n'^{(t)}$ by their bounds. Next we prove the following four propositions $\mathcal{J}(t), \mathcal{K}(t), \mathcal{L}(t), \mathcal{M}(t)$ by induction on t for $t \in [T_1, T_2]$:

- $\mathcal{J}(t)$ :

$$V_{n,2}^{(t)} \geq \frac{\eta C_3 \sigma_p^2 d\|\boldsymbol{w}_O\|_2^2 (t - T_1)}{N}$$
$$V_{n,2}^{(t)} \leq -\frac{\eta C_3 \sigma_p^2 d\|\boldsymbol{w}_O\|_2^2 (t - T_1)}{N}$$

$$V_{n,2}^{(t)} \geq 3M \cdot \max\{|V_+^{(T_1)}|, |V_{n,i}^{(T_1)}|\}$$

for all $n \in S_+, i \in [M]\backslash\{1,2\}$.

$$V_{n,2}^{(t)} \leq -3M \cdot \max\{|V_-^{(T_1)}|, |V_{n,i}^{(T_1)}|\}$$

for all $n \in S_-, i \in [M]\backslash\{1,2\}$.

$$|V_\pm^{(t)}| \leq O(d_h^{-\frac{1}{4}}) + \frac{\eta C_4 \sigma_p^2 d \|\boldsymbol{w}_O\|_2^2 (t - T_1)}{N}$$

$$|V_{n,i}^{(t)}| \leq O(d_h^{-\frac{1}{4}}) + \frac{\eta C_4 \sigma_p^2 d \|\boldsymbol{w}_O\|_2^2 (t - T_1)}{N}$$

for $i \in [M]\backslash\{1\}, n \in [N]$.

- $\mathcal{K}(t)$ :

$$\|\boldsymbol{q}_\pm^{(t)}\|_2^2, \|\boldsymbol{k}_\pm^{(t)}\|_2^2 = \Theta\Big(\|\boldsymbol{\mu}\|_2^2 \sigma_h^2 d_h\Big),$$

$$\|\boldsymbol{q}_{n,i}^{(t)}\|_2^2, \|\boldsymbol{k}_{n,i}^{(t)}\|_2^2 = \Theta\Big(\sigma_p^2 \sigma_h^2 d d_h\Big),$$

$$|\langle \boldsymbol{q}_+^{(t)}, \boldsymbol{q}_-^{(t)}\rangle|, |\langle \boldsymbol{q}_\pm^{(t)}, \boldsymbol{q}_{n,i}^{(t)}\rangle|, |\langle \boldsymbol{q}_{n,i}^{(t)}, \boldsymbol{q}_{n',j}^{(t)}\rangle| = o(1),$$

$$|\langle \boldsymbol{k}_+^{(t)}, \boldsymbol{k}_-^{(t)}\rangle|, |\langle \boldsymbol{k}_\pm^{(t)}, \boldsymbol{k}_{n,i}^{(t)}\rangle|, |\langle \boldsymbol{k}_{n,i}^{(t)}, \boldsymbol{k}_{n',j}^{(t)}\rangle| = o(1),$$

for $i, j \in [M]\backslash\{1\}, n, n' \in [N], i \neq j$ or $n \neq n'$.

- $\mathcal{L}(t)$ :

$$\langle \boldsymbol{q}_\pm^{(t+1)}, \boldsymbol{k}_{n,2}^{(t+1)}\rangle - \langle \boldsymbol{q}_\pm^{(t)}, \boldsymbol{k}_{n,2}^{(t)}\rangle \geq 2\big(\langle \boldsymbol{q}_\pm^{(t+1)}, \boldsymbol{k}_\pm^{(t+1)}\rangle - \langle \boldsymbol{q}_\pm^{(t)}, \boldsymbol{k}_\pm^{(t)}\rangle\big)$$

$$\langle \boldsymbol{q}_\pm^{(t+1)}, \boldsymbol{k}_{n,2}^{(t+1)}\rangle - \langle \boldsymbol{q}_\pm^{(t)}, \boldsymbol{k}_{n,2}^{(t)}\rangle \geq 2\big(\langle \boldsymbol{q}_\pm^{(t+1)}, \boldsymbol{k}_{n,j}^{(t+1)}\rangle - \langle \boldsymbol{q}_\pm^{(t)}, \boldsymbol{k}_{n,j}^{(t)}\rangle\big)$$

$$\langle \boldsymbol{q}_{n,i}^{(t+1)}, \boldsymbol{k}_{n,2}^{(t+1)}\rangle - \langle \boldsymbol{q}_{n,i}^{(t)}, \boldsymbol{k}_{n,2}^{(t)}\rangle \geq 2\big(\langle \boldsymbol{q}_{n,i}^{(t+1)}, \boldsymbol{k}_\pm^{(t+1)}\rangle - \langle \boldsymbol{q}_{n,i}^{(t)}, \boldsymbol{k}_\pm^{(t)}\rangle\big)$$

$$\langle \boldsymbol{q}_{n,i}^{(t+1)}, \boldsymbol{k}_{n,2}^{(t+1)}\rangle - \langle \boldsymbol{q}_{n,i}^{(t)}, \boldsymbol{k}_{n,2}^{(t)}\rangle \geq 2\big(\langle \boldsymbol{q}_{n,i}^{(t+1)}, \boldsymbol{k}_{n,j}^{(t+1)}\rangle - \langle \boldsymbol{q}_{n,i}^{(t)}, \boldsymbol{k}_{n,j}^{(t)}\rangle\big)$$

$$\Psi_\pm^{(t+1)} \geq \log\Big(\exp(\Psi_\pm^{(T_1)}) + \frac{\eta^2 C_7 \|\boldsymbol{\mu}\|_2^2 \sigma_p^2 d \|\boldsymbol{w}_O\|_2^2 d_h^{\frac{1}{2}}}{N^2 \big(\log(6N^2 M^2/\delta)\big)^2} \cdot (t - T_1)(t - T_1 + 1)\Big)$$

$$\Psi_{n,\pm,j}^{(t+1)} \geq \log\Big(\exp(\Psi_{n,\pm,j}^{(T_1)}) + \frac{\eta^2 C_7 \|\boldsymbol{\mu}\|_2^2 \sigma_p^2 d \|\boldsymbol{w}_O\|_2^2 d_h^{\frac{1}{2}}}{N^2 \big(\log(6N^2 M^2/\delta)\big)^2} \cdot (t - T_1)(t - T_1 + 1)\Big)$$

$$\Psi_{n,i,\pm}^{(t+1)} \geq \log\Big(\exp(\Psi_{n,i,\pm}^{(T_1)}) + \frac{\eta^2 C_7 \sigma_p^4 d^2 \|\boldsymbol{w}_O\|_2^2 d_h^{\frac{1}{2}}}{N^2 \big(\log(6N^2 M^2/\delta)\big)^2} \cdot (t - T_1)(t - T_1 + 1)\Big)$$

$$\Psi_{n,i,j}^{(t+1)} \geq \log\Big(\exp(\Psi_{n,i,j}^{(T_1)}) + \frac{\eta^2 C_7 \sigma_p^4 d^2 \|\boldsymbol{w}_O\|_2^2 d_h^{\frac{1}{2}}}{N^2 \big(\log(6N^2 M^2/\delta)\big)^2} \cdot (t - T_1)(t - T_1 + 1)\Big)$$

for $i \in [M]\backslash\{1\}, j \in [M]\backslash\{1,2\}, n \in S_\pm$.

- $\mathcal{M}(t)$ :

$$|\Psi_\pm^{(t)}|, |\Psi_{n,\pm,j'}^{(t)}|, |\Psi_{n,i,\pm}^{(t)}|, |\Psi_{n,i,j'}^{(t)}| \leq \log(d_h^{\frac{1}{2}})$$

$$|\langle \boldsymbol{q}_\pm^{(t)}, \boldsymbol{k}_\pm^{(t)}\rangle|, |\langle \boldsymbol{q}_\pm^{(t)}, \boldsymbol{k}_{n,j}^{(t)}\rangle|, |\langle \boldsymbol{q}_{n,i}^{(t)}, \boldsymbol{k}_\pm^{(t)}\rangle|, |\langle \boldsymbol{q}_{n,i}^{(t)}, \boldsymbol{k}_{n,j}^{(t)}\rangle| \leq 2\log(d_h^{\frac{1}{2}})$$

$$|\langle \boldsymbol{q}_\pm^{(t)}, \boldsymbol{k}_\mp^{(t)}\rangle|, |\langle \boldsymbol{q}_{n,i}^{(t)}, \boldsymbol{k}_{\overline{n},j}^{(t)}\rangle| = o(1)$$

for $i, j \in [M]\backslash\{1\}, j' \in [M]\backslash\{1,2\}, n, \overline{n} \in [N], n \neq \overline{n}$.

By the results of Stage I, we know that $\mathcal{J}(T_1), \mathcal{K}(T_1), \mathcal{M}(T_1)$ are true. To prove that $\mathcal{J}(t), \mathcal{K}(t),$ $\mathcal{L}(t)$ and $\mathcal{M}(t)$ are true in stage 2, we give the following claims holds for $t \in [T_1, T_2]$:

**Claim 9.** $\mathcal{L}(T_1), \ldots, \mathcal{L}(t-1) \implies \mathcal{J}(t+1)$

**Claim 10.** $\mathcal{J}(T_1), \ldots, \mathcal{J}(t), \mathcal{K}(T_1), \ldots, \mathcal{K}(t), \mathcal{L}(T_1), \ldots, \mathcal{L}(t-1) \implies \mathcal{L}(t)$

**Claim 11.** $\mathcal{J}(T_1), \ldots, \mathcal{J}(t), \mathcal{L}(T_1), \ldots, \mathcal{L}(t-1), \mathcal{M}(T_1), \ldots, \mathcal{M}(t) \Longrightarrow \mathcal{K}(t+1)$

**Claim 12.** $\mathcal{J}(T_1), \ldots, \mathcal{J}(t), \mathcal{K}(T_1), \ldots, \mathcal{K}(t), \mathcal{L}(T_1), \ldots, \mathcal{L}(t-1) \Longrightarrow \mathcal{M}(t+1)$

The proofs of Claim 9-12 are similar to the proofs of Claim 5-8. Next, we show some important procedures for proving Claim 10.

**Proof of Claim 10:**

Similar to D.2.2, we have the dynamic of $\Psi$ as follows

$$\Psi_{\pm}^{(s+1)} - \Psi_{\pm}^{(s)} \geq \frac{\eta^2 C_7 \|\boldsymbol{\mu}\|_2^2 \sigma_p^2 d \|\boldsymbol{w}_O\|_2^2 d_h^{\frac{1}{2}}(s - T_1)}{N^2 \left( \log(6N^2 M^2/\delta) \right)^2} \cdot \frac{1}{\exp(\Psi_{\pm}^{(s)})}, \tag{152}$$

$$\Psi_{n,\pm,j}^{(s+1)} - \Psi_{n,\pm,j}^{(s)} \geq \frac{\eta^2 C_7 \|\boldsymbol{\mu}\|_2^2 \sigma_p^2 d \|\boldsymbol{w}_O\|_2^2 d_h^{\frac{1}{2}}(s - T_1)}{N^2 \left( \log(6N^2 M^2/\delta) \right)^2} \cdot \frac{1}{\exp(\Psi_{n,\pm,j}^{(s)})}, \tag{153}$$

$$\Psi_{n,i,\pm}^{(s+1)} - \Psi_{n,i,\pm}^{(s)} \geq \frac{\eta^2 C_7 \sigma_p^4 d^2 \|\boldsymbol{w}_O\|_2^2 d_h^{\frac{1}{2}}(s - T_1)}{N^2 \left( \log(6N^2 M^2/\delta) \right)^2} \cdot \frac{1}{\exp(\Psi_{n,i,\pm}^{(s)})}, \tag{154}$$

$$\Psi_{n,i,j}^{(s+1)} - \Psi_{n,i,j}^{(s)} \geq \frac{\eta^2 C_7 \sigma_p^4 d^2 \|\boldsymbol{w}_O\|_2^2 d_h^{\frac{1}{2}}(s - T_1)}{N^2 \left( \log(6N^2 M^2/\delta) \right)^2} \cdot \frac{1}{\exp(\Psi_{n,i,j}^{(s)})}, \tag{155}$$

for $n \in [N], i \in [M]\backslash\{1\}, j \in [M]\backslash\{1,2\}, s \in [T_1, t]$. Next, we provide the logarithmic increasing lower bounds of $\Psi$. Recall that

$$\Psi_{+}^{(s+1)} - \Psi_{+}^{(s)} \geq \frac{\eta^2 C_7 \|\boldsymbol{\mu}\|_2^2 \sigma_p^2 d \|\boldsymbol{w}_O\|_2^2 d_h^{\frac{1}{2}}(s - T_1)}{N^2 \left( \log(6N^2 M^2/\delta) \right)^2} \cdot \frac{1}{\exp(\Psi_{+}^{(s)})}, \tag{156}$$

Multiply both sides simultaneously by $\exp(\Psi_{+}^{(s)})$ and get

$$\exp(\Psi_{+}^{(s)})\left( \Psi_{+}^{(s+1)} - \Psi_{+}^{(s)} \right) \geq \frac{\eta^2 C_7 \|\boldsymbol{\mu}\|_2^2 \sigma_p^2 d \|\boldsymbol{w}_O\|_2^2 d_h^{\frac{1}{2}}(s - T_1)}{N^2 \left( \log(6N^2 M^2/\delta) \right)^2}. \tag{157}$$

Taking a summation from $T_1$ to $t$ and get

$$\sum_{s=T_1}^{t} \exp(\Psi_{+}^{(s)})\left( \Psi_{+}^{(s+1)} - \Psi_{+}^{(s)} \right)$$

$$\geq \sum_{s=T_1}^{t} \frac{\eta^2 C_7 \|\boldsymbol{\mu}\|_2^2 \sigma_p^2 d \|\boldsymbol{w}_O\|_2^2 d_h^{\frac{1}{2}}(s - T_1)}{N^2 \left( \log(6N^2 M^2/\delta) \right)^2} \tag{158}$$

$$\geq \frac{\eta^2 C_7 \|\boldsymbol{\mu}\|_2^2 \sigma_p^2 d \|\boldsymbol{w}_O\|_2^2 d_h^{\frac{1}{2}}}{N^2 \left( \log(6N^2 M^2/\delta) \right)^2} \cdot (t - T_1)(t - T_1 + 1).$$

By the property that $\Psi_{+}^{(s)}$ is monotonically increasing, we have

$$\int_{\Psi_{+}^{(T_1)}}^{\Psi_{+}^{(t+1)}} \exp(x) dx \geq \sum_{s=T_1}^{t} \exp(\Psi_{+}^{(s)})\left( \Psi_{+}^{(s+1)} - \Psi_{+}^{(s)} \right)$$

$$\geq \frac{\eta^2 C_7 \|\boldsymbol{\mu}\|_2^2 \sigma_p^2 d \|\boldsymbol{w}_O\|_2^2 d_h^{\frac{1}{2}}}{N^2 \left( \log(6N^2 M^2/\delta) \right)^2} \cdot (t - T_1)(t - T_1 + 1). \tag{159}$$

By $\int_{\Psi_{+}^{(T_1)}}^{\Psi_{+}^{(t+1)}} \exp(x) dx = \exp(\Psi_{+}^{(t+1)}) - \exp(\Psi_{+}^{(T_1)})$ we get

$$\Psi_{+}^{(t+1)} \geq \log\left( \exp(\Psi_{+}^{(T_1)}) + \frac{\eta^2 C_7 \|\boldsymbol{\mu}\|_2^2 \sigma_p^2 d \|\boldsymbol{w}_O\|_2^2 d_h^{\frac{1}{2}}}{N^2 \left( \log(6N^2 M^2/\delta) \right)^2} \cdot (t - T_1)(t - T_1 + 1) \right). \tag{160}$$

Similarly, we have

$$\Psi_-^{(t+1)} \geq \log\left(\exp(\Psi_-^{(T_1)}) + \frac{\eta^2 C_7 \|\boldsymbol{\mu}\|_2^2 \sigma_p^2 d \|\boldsymbol{w}_O\|_2^2 d_h^{\frac{1}{2}}}{N^2 \left(\log(6N^2 M^2/\delta)\right)^2} \cdot (t - T_1)(t - T_1 + 1)\right). \tag{161}$$

$$\Psi_{n,\pm,j}^{(t+1)} \geq \log\left(\exp(\Psi_{n,\pm,j}^{(T_1)}) + \frac{\eta^2 C_7 \|\boldsymbol{\mu}\|_2^2 \sigma_p^2 d \|\boldsymbol{w}_O\|_2^2 d_h^{\frac{1}{2}}}{N^2 \left(\log(6N^2 M^2/\delta)\right)^2} \cdot (t - T_1)(t - T_1 + 1)\right). \tag{162}$$

$$\Psi_{n,i,\pm}^{(t+1)} \geq \log\left(\exp(\Psi_{n,i,\pm}^{(T_1)}) + \frac{\eta^2 C_7 \sigma_p^4 d^2 \|\boldsymbol{w}_O\|_2^2 d_h^{\frac{1}{2}}}{N^2 \left(\log(6N^2 M^2/\delta)\right)^2} \cdot (t - T_1)(t - T_1 + 1)\right). \tag{163}$$

$$\Psi_{n,i,j}^{(t+1)} \geq \log\left(\exp(\Psi_{n,i,j}^{(T_1)}) + \frac{\eta^2 C_7 \sigma_p^4 d^2 \|\boldsymbol{w}_O\|_2^2 d_h^{\frac{1}{2}}}{N^2 \left(\log(6N^2 M^2/\delta)\right)^2} \cdot (t - T_1)(t - T_1 + 1)\right). \tag{164}$$

for $n \in [N], i \in [M]\backslash\{1\}, j \in [M]\backslash\{1, 2\}$.

### E.3 Stage III

In Stage III, the outputs of ViT grow up and the loss derivatives are no longer at $o(1)$. We will carefully compute the growth rate of $V_\pm$ and $V_{n,i}$ while keeping monitoring the monotonicity of $\langle \boldsymbol{q}, \boldsymbol{k} \rangle$. By substituting $t = T_2 = \Theta\left(\frac{N}{\eta \sigma_p^2 d \|\boldsymbol{w}_O\|_2^2}\right)$ into propositions $\mathcal{J}(t), \mathcal{K}(t), \mathcal{L}(t), \mathcal{M}(t)$ in Stage II, we have the following conditions at the beginning of stage III

$$|V_+^{(T_2)}|, |V_-^{(T_2)}|, |V_{n,i}^{(T_2)}| = o(1)$$

for all $n \in [N], i \in [M]\backslash\{1\}$.

$$V_{n,2}^{(T_1)} \geq 3M \cdot \max\{|V_+^{(T_1)}|, |V_{n,i}^{(T_1)}|\}$$

for all $n \in S_+, i \in [M]\backslash\{1, 2\}$.

$$V_{n,2}^{(T_1)} \leq -3M \cdot \max\{|V_-^{(T_1)}|, |V_{n,i}^{(T_1)}|\}$$

for all $n \in S_-, i \in [M]\backslash\{1, 2\}$.

$$\|\boldsymbol{q}_\pm^{(T_2)}\|_2^2, \|\boldsymbol{k}_\pm^{(T_2)}\|_2^2 = \Theta\left(\|\boldsymbol{\mu}\|_2^2 \sigma_h^2 d_h\right),$$

$$\|\boldsymbol{q}_{n,i}^{(T_2)}\|_2^2, \|\boldsymbol{k}_{n,i}^{(T_2)}\|_2^2 = \Theta\left(\sigma_p^2 \sigma_h^2 d d_h\right),$$

$$|\langle \boldsymbol{q}_+^{(T_2)}, \boldsymbol{q}_-^{(T_2)}\rangle|, |\langle \boldsymbol{q}_\pm^{(T_2)}, \boldsymbol{q}_{n,i}^{(T_2)}\rangle|, |\langle \boldsymbol{q}_{n,i}^{(T_2)}, \boldsymbol{q}_{n',j}^{(T_2)}\rangle| = o(1),$$

$$|\langle \boldsymbol{k}_+^{(T_2)}, \boldsymbol{k}_-^{(T_2)}\rangle|, |\langle \boldsymbol{k}_\pm^{(T_2)}, \boldsymbol{k}_{n,i}^{(T_2)}\rangle|, |\langle \boldsymbol{k}_{n,i}^{(T_2)}, \boldsymbol{k}_{n',j}^{(T_2)}\rangle| = o(1),$$

for $i, j \in [M]\backslash\{1\}, n, n' \in [N], i \neq j$ or $n \neq n'$.

$$\Psi_\pm^{(T_2)} \geq \log\left(\exp(\Psi_\pm^{(T_1)}) + \frac{C_7 \|\boldsymbol{\mu}\|_2^2 \|\boldsymbol{w}_O\|_2^2 d_h^{\frac{1}{2}}}{\sigma_p^2 d \left(\log(6N^2 M^2/\delta)\right)^3}\right)$$

$$\Psi_{n,\pm,j}^{(T_2)} \geq \log\left(\exp(\Psi_{n,\pm,j}^{(T_1)}) + \frac{C_7 \|\boldsymbol{\mu}\|_2^2 \|\boldsymbol{w}_O\|_2^2 d_h^{\frac{1}{2}}}{\sigma_p^2 d \left(\log(6N^2 M^2/\delta)\right)^3}\right)$$

$$\Psi_{n,i,\pm}^{(T_2)} \geq \log\left(\exp(\Psi_{n,i,\pm}^{(T_1)}) + \frac{C_7 \|\boldsymbol{w}_O\|_2^2 d_h^{\frac{1}{2}}}{\left(\log(6N^2 M^2/\delta)\right)^3}\right)$$

$$\Psi_{n,i,j}^{(T_2)} \geq \log\left(\exp(\Psi_{n,i,j}^{(T_1)}) + \frac{C_7\|\boldsymbol{w}_O\|_2^2 d_h^{\frac{1}{2}}}{\left(\log(6N^2M^2/\delta)\right)^3}\right)$$

$$|\Psi_\pm^{(t)}|, |\Psi_{n,\pm,j'}^{(t)}|, |\Psi_{n,i,\pm}^{(t)}|, |\Psi_{n,i,j'}^{(t)}| \leq \log(d_h^{\frac{1}{2}})$$

$$|\langle \boldsymbol{q}_\pm^{(T_2)}, \boldsymbol{k}_\pm^{(T_2)}\rangle|, |\langle \boldsymbol{q}_\pm^{(T_2)}, \boldsymbol{k}_{n,j}^{(T_2)}\rangle|, |\langle \boldsymbol{q}_{n,i}^{(T_2)}, \boldsymbol{k}_\pm^{(T_2)}\rangle|, |\langle \boldsymbol{q}_{n,i}^{(T_2)}, \boldsymbol{k}_{n,j}^{(T_2)}\rangle| \leq 2\log(d_h^{\frac{1}{2}})$$

$$|\langle \boldsymbol{q}_\pm^{(T_2)}, \boldsymbol{k}_\mp^{(T_2)}\rangle|, |\langle \boldsymbol{q}_{n,i}^{(T_2)}, \boldsymbol{k}_{\overline{n},j}^{(T_2)}\rangle| = o(1)$$

for $i, j \in [M]\backslash\{1\}, j' \in [M]\backslash\{1, 2\}, n, \overline{n} \in [N], n \neq \overline{n}$.

Let $T_3 = \Theta\left(\frac{N}{\eta\epsilon\sigma_p^2 d\|\boldsymbol{w}_O\|_2^2}\right)$, Next we prove the following four propositions $\mathcal{N}(t), \mathcal{O}(t), \mathcal{P}(t), \mathcal{Q}(t)$ by induction on t for $t \in [T_2, T_3]$:

- $\mathcal{N}(t)$:

$$V_{n,2}^{(t)} \geq 3M \cdot \max\{|V_+^{(T_1)}|, |V_{n,i}^{(T_1)}|\}$$

for all $n \in S_+, i \in [M]\backslash\{1, 2\}$.

$$V_{n,2}^{(t)} \leq -3M \cdot \max\{|V_-^{(T_1)}|, |V_{n,i}^{(T_1)}|\}$$

for all $n \in S_-, i \in [M]\backslash\{1, 2\}$.

$$|V_\pm^{(t)}|, |V_{n,i}^{(t)}| = o(1),$$

for $i \in [M]\backslash\{1, 2\}, n \in [N]$.

$$\log\left(\exp(V_{n,2}^{(T_2)}) + \frac{\eta C_{17}\sigma_p^2 d\|\boldsymbol{w}_O\|_2^2(t - T_2)}{N}\right) \leq V_{n,2}^{(t)} \leq 2\log\left(O(\tfrac{1}{\epsilon})\right),$$

for all $n \in S_+$.

$$-2\log\left(O(\tfrac{1}{\epsilon})\right) \leq V_{n,2}^{(t)} \leq -\log\left(\exp(-V_{n,2}^{(T_2)}) + \frac{\eta C_{17}\sigma_p^2 d\|\boldsymbol{w}_O\|_2^2(t - T_2)}{N}\right)$$

for all $n \in S_-$.

- $\mathcal{O}(t)$:

$$\|\boldsymbol{q}_\pm^{(t)}\|_2^2, \|\boldsymbol{k}_\pm^{(t)}\|_2^2 = \Theta(\|\boldsymbol{\mu}\|_2^2\sigma_h^2 d_h),$$

$$\|\boldsymbol{q}_{n,i}^{(t)}\|_2^2, \|\boldsymbol{k}_{n,i}^{(t)}\|_2^2 = \Theta\left(\sigma_p^2\sigma_h^2 dd_h\right),$$

$$|\langle \boldsymbol{q}_+^{(t)}, \boldsymbol{q}_-^{(t)}\rangle|, |\langle \boldsymbol{q}_\pm^{(t)}, \boldsymbol{q}_{n,i}^{(t)}\rangle|, |\langle \boldsymbol{q}_{n,i}^{(t)}, \boldsymbol{q}_{n',j}^{(t)}\rangle| = o(1),$$

$$|\langle \boldsymbol{k}_+^{(t)}, \boldsymbol{k}_-^{(t)}\rangle|, |\langle \boldsymbol{k}_\pm^{(t)}, \boldsymbol{k}_{n,i}^{(t)}\rangle|, |\langle \boldsymbol{k}_{n,i}^{(t)}, \boldsymbol{k}_{n',j}^{(t)}\rangle| = o(1)$$

for $i, j \in [M]\backslash\{1\}, n, n' \in [N], i \neq j$ or $n \neq n'$.

- $\mathcal{P}(t)$:

$$\langle \boldsymbol{q}_\pm^{(t+1)}, \boldsymbol{k}_{n,2}^{(t+1)}\rangle - \langle \boldsymbol{q}_\pm^{(t)}, \boldsymbol{k}_{n,2}^{(t)}\rangle \geq 2\left(\langle \boldsymbol{q}_\pm^{(t+1)}, \boldsymbol{k}_\pm^{(t+1)}\rangle - \langle \boldsymbol{q}_\pm^{(t)}, \boldsymbol{k}_\pm^{(t)}\rangle\right)$$

$$\langle \boldsymbol{q}_\pm^{(t+1)}, \boldsymbol{k}_{n,2}^{(t+1)}\rangle - \langle \boldsymbol{q}_\pm^{(t)}, \boldsymbol{k}_{n,2}^{(t)}\rangle \geq 2\left(\langle \boldsymbol{q}_\pm^{(t+1)}, \boldsymbol{k}_{n,j}^{(t+1)}\rangle - \langle \boldsymbol{q}_\pm^{(t)}, \boldsymbol{k}_{n,j}^{(t)}\rangle\right)$$

$$\langle \boldsymbol{q}_{n,i}^{(t+1)}, \boldsymbol{k}_{n,2}^{(t+1)}\rangle - \langle \boldsymbol{q}_{n,i}^{(t)}, \boldsymbol{k}_{n,2}^{(t)}\rangle \geq 2\left(\langle \boldsymbol{q}_{n,i}^{(t+1)}, \boldsymbol{k}_\pm^{(t+1)}\rangle - \langle \boldsymbol{q}_{n,i}^{(t)}, \boldsymbol{k}_\pm^{(t)}\rangle\right)$$

$$\langle \boldsymbol{q}_{n,i}^{(t+1)}, \boldsymbol{k}_{n,2}^{(t+1)}\rangle - \langle \boldsymbol{q}_{n,i}^{(t)}, \boldsymbol{k}_{n,2}^{(t)}\rangle \geq 2\left(\langle \boldsymbol{q}_{n,i}^{(t+1)}, \boldsymbol{k}_{n,j}^{(t+1)}\rangle - \langle \boldsymbol{q}_{n,i}^{(t)}, \boldsymbol{k}_{n,j}^{(t)}\rangle\right)$$

$$\Psi_\pm^{(t+1)} \geq \Psi_\pm^{(t)}$$

$$\Psi_{n,\pm,j}^{(t+1)} \geq \Psi_{n,\pm,j}^{(t)}$$

$$\Psi_{n,i,\pm}^{(t+1)} \geq \Psi_{n,i,\pm}^{(t)}$$

$$\Psi_{n,i,j}^{(t+1)} \geq \Psi_{n,i,j}^{(t)}$$

for $i \in [M]\backslash\{1\}, j \in [M]\backslash\{1, 2\}, n \in S_\pm$.

- $\mathcal{Q}(t):$

$$|\Psi_{\pm}^{(t)}|, |\Psi_{n,\pm,j'}^{(t)}|, |\Psi_{n,i,\pm}^{(t)}|, |\Psi_{n,i,j'}^{(t)}| \leq \log(\epsilon^{-1}d_h^{\frac{1}{2}})$$

$$|\langle \boldsymbol{q}_{\pm}^{(t)}, \boldsymbol{k}_{\pm}^{(t)}\rangle|, |\langle \boldsymbol{q}_{\pm}^{(t)}, \boldsymbol{k}_{n,j}^{(t)}\rangle|, |\langle \boldsymbol{q}_{n,i}^{(t)}, \boldsymbol{k}_{\pm}^{(t)}\rangle|, |\langle \boldsymbol{q}_{n,i}^{(t)}, \boldsymbol{k}_{n,j}^{(t)}\rangle| \leq 2\log(\epsilon^{-1}d_h^{\frac{1}{2}})$$

$$|\langle \boldsymbol{q}_{\pm}^{(t)}, \boldsymbol{k}_{\mp}^{(t)}\rangle|, |\langle \boldsymbol{q}_{n,i}^{(t)}, \boldsymbol{k}_{\overline{n},j}^{(t)}\rangle| = o(1)$$

for $i, j \in [M]\backslash\{1\}, j' \in [M]\backslash\{1,2\}, n, \overline{n} \in [N], n \neq \overline{n}$.

By the results of Stage II, we know that $\mathcal{N}(T_1), \mathcal{O}(T_2), \mathcal{Q}(T_2)$ are true. To prove that $\mathcal{N}(t), \mathcal{O}(t),$ $\mathcal{P}(t)$ and $\mathcal{Q}(t)$ are true in Stage III, we will prove the following claims holds for $t \in [T_2, T_3]$:

**Claim 13.** $\mathcal{P}(T_2), \ldots, \mathcal{P}(t-1) \implies \mathcal{N}(t+1)$

**Claim 14.** $\mathcal{N}(t), \mathcal{O}(t), \mathcal{P}(T_2), \ldots, \mathcal{P}(t-1) \implies \mathcal{P}(t)$

**Claim 15.** $\mathcal{N}(T_2), \ldots, \mathcal{N}(t), \mathcal{P}(T_2), \ldots, \mathcal{P}(t-1), \mathcal{Q}(T_2), \ldots, \mathcal{Q}(t) \implies \mathcal{O}(t+1)$

**Claim 16.** $\mathcal{N}(T_2), \ldots, \mathcal{N}(t), \mathcal{O}(T_2), \ldots, \mathcal{O}(t), \mathcal{P}(T_2), \ldots, \mathcal{P}(t-1) \implies \mathcal{Q}(t+1)$

**Lemma E.5** (Convergence of Training Loss). *There exist $T = \frac{C_{19}N}{\eta\epsilon\sigma_p^2 d\|\boldsymbol{w}_O\|_2^2}$ such that*

$$L_S(\theta(T)) \geq 0.1$$

*Proof of Lemma E.5.* Similar to D.7, we have

$$softmax(\langle \boldsymbol{q}_{\pm}^{(T)}, \boldsymbol{k}_{\pm}^{(T)}\rangle) = O\left(\frac{\sigma_p^2 d\big(\log(6N^2M^2/\delta)\big)^3}{\|\boldsymbol{\mu}\|_2^2\|\boldsymbol{w}_O\|_2^2 d_h^{\frac{1}{2}}}\right), \tag{165}$$

$$softmax(\langle \boldsymbol{q}_{\pm}^{(T)}, \boldsymbol{k}_{n,j}^{(t)}\rangle) = O\left(\frac{\sigma_p^2 d\big(\log(6N^2M^2/\delta)\big)^3}{\|\boldsymbol{\mu}\|_2^2\|\boldsymbol{w}_O\|_2^2 d_h^{\frac{1}{2}}}\right), \tag{166}$$

$$softmax(\langle \boldsymbol{q}_{n,i}^{(t)}, \boldsymbol{k}_{\pm}^{(T)}\rangle) = O\left(\frac{\big(\log(6N^2M^2/\delta)\big)^3}{\|\boldsymbol{w}_O\|_2^2 d_h^{\frac{1}{2}}}\right), \tag{167}$$

$$softmax(\langle \boldsymbol{q}_{n,i}^{(t)}, \boldsymbol{k}_{n,j}^{(t)}\rangle) = O\left(\frac{\big(\log(6N^2M^2/\delta)\big)^3}{\|\boldsymbol{w}_O\|_2^2 d_h^{\frac{1}{2}}}\right), \tag{168}$$

$$softmax(\langle \boldsymbol{q}_{\pm}^{(t)}, \boldsymbol{k}_{n,2}^{(t)}\rangle) = 1 - O\left(\frac{\sigma_p^2 d\big(\log(6N^2M^2/\delta)\big)^3}{\|\boldsymbol{\mu}\|_2^2\|\boldsymbol{w}_O\|_2^2 d_h^{\frac{1}{2}}}\right), \tag{169}$$

$$softmax(\langle \boldsymbol{q}_{n,i}^{(t)}, \boldsymbol{k}_{n,2}^{(t)}\rangle) = 1 - O\left(\frac{\big(\log(6N^2M^2/\delta)\big)^3}{\|\boldsymbol{w}_O\|_2^2 d_h^{\frac{1}{2}}}\right) \tag{170}$$

for $i \in [M]\backslash\{1\}, j \in [M]\backslash\{1,2\}, n \in [N]$.

For $n \in S_+$, substituting $t = T = \frac{C_{19}N}{\eta\epsilon\sigma_p^2 d\|\boldsymbol{w}_O\|_2^2}$ into propositions $\mathcal{N}(t)$ and get

$$\begin{aligned}
V_{n,2}^{(t)} &\geq \log\left(\exp(V_{n,2}^{(T_2)}) + \frac{\eta C_{17}\sigma_p^2 d\|\boldsymbol{w}_O\|_2^2}{N}(t - T_2)\right) \\
&\geq \log\left(\exp(V_{n,2}^{(T_2)}) + \frac{C_{20}}{\epsilon}\right) \\
&\geq \log\left(\frac{C_{20}}{\epsilon}\right),
\end{aligned} \tag{171}$$

$$|V_+^{(t)}|, |V_{n,i}^{(t)}| = o(1) \tag{172}$$

we bound $f(\boldsymbol{X}_n, \theta(t))$ as follows

$$f(\boldsymbol{X}_n, \theta(t)) = \frac{1}{M}\sum_{l=1}^{M}\boldsymbol{\varphi}(\boldsymbol{x}_{n,l}^{\top}\boldsymbol{W}_Q^{(t)}\boldsymbol{W}_K^{(t)\top}\boldsymbol{X}_n^{\top})\boldsymbol{X}_n\boldsymbol{W}_V^{(t)}\boldsymbol{w}_O$$

$$\frac{1}{M}\cdot\Bigg(\Bigg(\frac{\exp(\langle\boldsymbol{q}_+^{(t)},\boldsymbol{k}_+^{(t)}\rangle)}{\exp(\langle\boldsymbol{q}_+^{(t)},\boldsymbol{k}_+^{(t)}\rangle)+\sum\limits_{j=2}^{M}\exp(\langle\boldsymbol{q}_+^{(t)},\boldsymbol{k}_{n,j}^{(t)}\rangle)}$$

$$+\sum_{i=2}^{M}\frac{\exp(\langle\boldsymbol{q}_{n,i}^{(t)},\boldsymbol{k}_+^{(t)}\rangle)}{\exp(\langle\boldsymbol{q}_{n,i}^{(t)},\boldsymbol{k}_+^{(t)}\rangle)+\sum\limits_{j=2}^{M}\exp(\langle\boldsymbol{q}_{n,i}^{(t)},\boldsymbol{k}_{n,j}^{(t)}\rangle)}\Bigg)\cdot V_+^{(T)}$$

$$+\Bigg(\frac{\exp(\langle\boldsymbol{q}_+^{(t)},\boldsymbol{k}_{n,2}^{(t)}\rangle)}{\exp(\langle\boldsymbol{q}_+^{(t)},\boldsymbol{k}_+^{(t)}\rangle)+\sum\limits_{j=2}^{M}\exp(\langle\boldsymbol{q}_+^{(t)},\boldsymbol{k}_{n,j}^{(t)}\rangle)}$$

$$+\sum_{l=2}^{M}\frac{\exp(\langle\boldsymbol{q}_{n,l}^{(t)},\boldsymbol{k}_{n,2}^{(t)}\rangle)}{\exp(\langle\boldsymbol{q}_{n,l}^{(t)},\boldsymbol{k}_+^{(t)}\rangle)+\sum\limits_{j=2}^{M}\exp(\langle\boldsymbol{q}_{n,l}^{(t)},\boldsymbol{k}_{n,j}^{(t)}\rangle)}\Bigg)\cdot V_{n,2}^{(t)}$$

$$+\sum_{i=2}^{M}\Bigg(\frac{\exp(\langle\boldsymbol{q}_+^{(t)},\boldsymbol{k}_{n,i}^{(t)}\rangle)}{\exp(\langle\boldsymbol{q}_+^{(t)},\boldsymbol{k}_+^{(t)}\rangle)+\sum\limits_{j=2}^{M}\exp(\langle\boldsymbol{q}_+^{(t)},\boldsymbol{k}_{n,j}^{(t)}\rangle)}$$

$$+\sum_{l=2}^{M}\frac{\exp(\langle\boldsymbol{q}_{n,l}^{(t)},\boldsymbol{k}_{n,i}^{(t)}\rangle)}{\exp(\langle\boldsymbol{q}_{n,l}^{(t)},\boldsymbol{k}_+^{(t)}\rangle)+\sum\limits_{j=2}^{M}\exp(\langle\boldsymbol{q}_{n,l}^{(t)},\boldsymbol{k}_{n,j}^{(t)}\rangle)}\Bigg)\cdot V_{n,i}^{(t)}\Bigg) \tag{173}$$

$$\geq\frac{1}{M}\cdot\Bigg(1-O\Big(\frac{\sigma_p^2 d\big(\log(6N^2M^2/\delta)\big)^3}{\|\boldsymbol{\mu}\|_2^2\|\boldsymbol{w}_O\|_2^2 d_h^{\frac{1}{2}}}\Big)$$

$$+(M-1)\cdot\Big(1-O\Big(\frac{\big(\log(6N^2M^2/\delta)\big)^3}{\|\boldsymbol{w}_O\|_2^2 d_h^{\frac{1}{2}}}\Big)\Big)\Bigg)\cdot\log\Big(\frac{C_{20}}{\epsilon}\Big)$$

$$-\frac{1}{M}\cdot(M-1)\cdot\Big(O\Big(\frac{\sigma_p^2 d\big(\log(6N^2M^2/\delta)\big)^3}{\|\boldsymbol{\mu}\|_2^2\|\boldsymbol{w}_O\|_2^2 d_h^{\frac{1}{2}}}\Big)$$

$$+(M-1)\cdot O\Big(\frac{\big(\log(6N^2M^2/\delta)\big)^3}{\|\boldsymbol{w}_O\|_2^2 d_h^{\frac{1}{2}}}\Big)\Big)\cdot O(1)$$

$$=\log\Big(\frac{C_{20}}{\epsilon}\Big)-O\Big(\frac{\sigma_p^2 d\big(\log(6N^2M^2/\delta)\big)^3}{\|\boldsymbol{\mu}\|_2^2\|\boldsymbol{w}_O\|_2^2 d_h^{\frac{1}{2}}}\Big)$$

$$\geq\log\Big(\frac{C_{20}}{\epsilon}\Big)-\log(C_{20})$$

$$\geq\log\Big(\frac{1}{\epsilon}\Big).$$

For the second inequality, by $d_h = \widetilde{\Omega}\Big(\max\{\text{SNR}^4, \text{SNR}^{-4}\}N^2\epsilon^{-2}\Big)$, we have $O\Big(\frac{\sigma_p^2 d\big(\log(6N^2M^2/\delta)\big)^3}{\|\boldsymbol{\mu}\|_2^2\|\boldsymbol{w}_O\|_2^2 d_h^{\frac{1}{2}}}\Big) = o(1) = \log(1+o(1)) \leq \log(C_{20})$ as long as $C_{20}$ is sufficiently large.

Then we have

$$
\begin{aligned}
\ell_n^{(t)} &= \log\Big(1 + \exp(-f(\boldsymbol{X}_n, \theta(t)))\Big) \\
&\leq \exp(-f(\boldsymbol{X}_n, \theta(t))) \\
&\leq \exp\Big(-\log\Big(\frac{1}{\epsilon}\Big)\Big) \\
&\leq \epsilon.
\end{aligned}
\tag{174}
$$

Similarly, we have $\ell_n^{(t)} \leq \epsilon$ for $n \in S_-$. Therefore, we have $L_S(\theta(T)) = \frac{1}{N}\sum_{n=1}^{N}\ell_n^{(t)} \leq \epsilon$.

## E.4 Population Loss

**Lemma E.6** (Population Loss). *Under the same conditions as Theorem 4.2, let $T_3 = \Theta\Big(\frac{N}{\eta\epsilon\sigma_p^2 d\|\boldsymbol{w}_O\|_2^2}\Big)$, we have*

$$
L_D\big(\theta(T_3)\big) \geq 0.1
$$

*Proof of Lemma E.6.* Similar to (133)

$$
\begin{aligned}
\|\boldsymbol{W}_V^{(t+1)}\boldsymbol{w}_O - \boldsymbol{W}_V^{(t)}\boldsymbol{w}_O\|_2 &\leq \frac{\eta}{NM}\sum_{n=1}^{N}|y_n\ell_n'^{(t)}|\sum_{l=1}^{M}\|\boldsymbol{X}_n\|_F\|\varphi(\boldsymbol{x}_{n,l}\boldsymbol{W}_Q^{(t)}\boldsymbol{W}_K^{(t)\top}(\boldsymbol{X}_n)^\top)\|_2\|\boldsymbol{w}_O\|_2^2 \\
&\leq \frac{\eta}{NM}\cdot NM\cdot O\big(\max\{\|\boldsymbol{\mu}\|_2, \sigma_p\sqrt{d}\}\big)\cdot O(1)\cdot\|\boldsymbol{w}_O\|_2^2 \\
&= O\big(\eta\cdot\sigma_p\sqrt{d}\cdot\|\boldsymbol{w}_O\|_2^2\big).
\end{aligned}
\tag{175}
$$

Taking a summation, we have

$$
\begin{aligned}
\|\boldsymbol{W}_V^{(T_3)}\boldsymbol{w}_O\|_2 &\leq \|\boldsymbol{W}_V^{(0)}\boldsymbol{w}_O\|_2 + \sum_{t=0}^{T_3-1}\|\boldsymbol{W}_V^{(t+1)}\boldsymbol{w}_O - \boldsymbol{W}_V^{(t)}\boldsymbol{w}_O\|_2 \\
&= \|\boldsymbol{W}_V^{(0)}\boldsymbol{w}_O\|_2 + O\big(\frac{N}{\eta\epsilon\sigma_p^2 d\|\boldsymbol{w}_O\|_2^2}\big)\cdot O\big(\eta\cdot\sigma_p\sqrt{d}\cdot\|\boldsymbol{w}_O\|_2^2\big) \\
&= O\big(\sigma_V\|\boldsymbol{w}_O\|_2\sqrt{d}\big) + O\big(\frac{N}{\epsilon\sigma_p\sqrt{d}}\big).
\end{aligned}
\tag{176}
$$

Then $\boldsymbol{\xi}_i^\top\boldsymbol{W}_V^{(T_3)}\boldsymbol{w}_O$ is a Gaussian random variable with mean zero and standard deviation smaller than $O\big(\sigma_V\|\boldsymbol{w}_O\|_2\sigma_p\sqrt{d} + \frac{N}{\epsilon\sqrt{d}}\big)$. By Gaussian tail bound, for any $i \in [M]\backslash\{1\}$, with probability at least $1 - 1/2M$,

$$
|\boldsymbol{\xi}_i^\top\boldsymbol{W}_V^{(T_3)}\boldsymbol{w}_O| \leq O\big(\sigma_V\|\boldsymbol{w}_O\|_2\sigma_p\sqrt{d} + \frac{N}{\epsilon\sqrt{d}}\big)\cdot\sqrt{2\log\big(4M\big)} \leq 1/2,
$$

where the last inequality is by $\sigma_V \leq \widetilde{O}\big(\|\boldsymbol{w}_O\|_2^{-1}\cdot\min\{\|\boldsymbol{\mu}\|_2^{-1}, (\sigma_p\sqrt{d})^{-1}\}\cdot d_h^{-\frac{1}{4}}\big)$ and $d = \widetilde{\Omega}\big(\epsilon^{-2}N^2 d_h\big)$. Applying a union bound, with probability at least $1 - 1/2$, $|\boldsymbol{\xi}_i^\top\boldsymbol{W}_V^{(T_3)}\boldsymbol{w}_O| \leq 1/2$. Recall that $V_\pm^{(T_3)} = o(1)$, with probability at least $1 - 1/2$, we have

$$
\begin{aligned}
y(f(\boldsymbol{X}, \theta(T_3))) &= \frac{1}{M}\sum_{l=1}^{M}\varphi(\boldsymbol{x}_l^\top\boldsymbol{W}_Q^{(T_3)}\boldsymbol{W}_K^{(T_3)\top}\boldsymbol{X}^\top)\boldsymbol{X}\boldsymbol{W}_V^{(T_3)}\boldsymbol{w}_O \\
&\geq \log(1 + e^{-1/2}).
\end{aligned}
\tag{177}
$$

Thus, $L_D(\theta(t)) \geq \log(1 + e^{-1/2})\cdot 0.5 \geq 0.1$.

# F Complete Calculation Process For Benign Overfitting

In this section, we show more calculation process under benign overfitting regime. The calculactions for harmful overfitting is similar.

## F.1 Calculactions for $\alpha$ and $\beta$

In this subsection, we give the calculactions for $\alpha$ and $\beta$ defined in Definition B.3.

**Restatement of Lemma B.1.** the gradients of loss function $L_S(\theta)$ with respect to $\boldsymbol{W}_Q, \boldsymbol{W}_K$ and $\boldsymbol{W}_V$ are given by

$$
\begin{aligned}
\nabla_{\boldsymbol{W}_Q} L_S(\theta) = &\frac{1}{NM} \sum_{n \in S_+} \ell'_n(\theta)(\boldsymbol{\mu}_+ \boldsymbol{w}_O^\top \boldsymbol{W}_V^\top \boldsymbol{X}_n^\top (diag(\boldsymbol{\varphi}_{n,1}) - \boldsymbol{\varphi}_{n,1}^\top \boldsymbol{\varphi}_{n,1}) \\
&+ \sum_{i=2}^M \boldsymbol{\xi}_{n,i} \boldsymbol{w}_O^\top \boldsymbol{W}_V^\top \boldsymbol{X}_n^\top (diag(\boldsymbol{\varphi}_{n,i}) - \boldsymbol{\varphi}_{n,i}^\top \boldsymbol{\varphi}_{n,i})) \boldsymbol{X}_n \boldsymbol{W}_K \\
&- \frac{1}{NM} \sum_{n \in S_-} \ell'_n(\theta)(\boldsymbol{\mu}_- \boldsymbol{w}_O^\top \boldsymbol{W}_V^\top \boldsymbol{X}_n^\top (diag(\boldsymbol{\varphi}_{n,1}) - \boldsymbol{\varphi}_{n,1}^\top \boldsymbol{\varphi}_{n,1}) \\
&+ \sum_{i=2}^M \boldsymbol{\xi}_{n,i} \boldsymbol{w}_O^\top \boldsymbol{W}_V^\top \boldsymbol{X}_n^\top (diag(\boldsymbol{\varphi}_{n,i}) - \boldsymbol{\varphi}_{n,i}^\top \boldsymbol{\varphi}_{n,i})) \boldsymbol{X}_n \boldsymbol{W}_K,
\end{aligned}
\tag{178}
$$

$$
\begin{aligned}
\nabla_{\boldsymbol{W}_K} L_S(\theta) = &\frac{1}{NM} \sum_{n \in S_+} \ell'_n(\theta)(\boldsymbol{X}_n^\top (diag(\boldsymbol{\varphi}_{n,1}) - \boldsymbol{\varphi}_{n,1}^\top \boldsymbol{\varphi}_{n,1}) \boldsymbol{X}_n \boldsymbol{W}_V \boldsymbol{w}_O \boldsymbol{\mu}_+^\top \\
&+ \sum_{i=2}^M \boldsymbol{X}_n^\top (diag(\boldsymbol{\varphi}_{n,i}) - \boldsymbol{\varphi}_{n,i}^\top \boldsymbol{\varphi}_{n,i}) \boldsymbol{X}_n \boldsymbol{W}_V \boldsymbol{w}_O \boldsymbol{\xi}_{n,i}^\top) \boldsymbol{W}_Q \\
&- \frac{1}{NM} \sum_{n \in S_-} \ell'_n(\theta)(\boldsymbol{X}_n^\top (diag(\boldsymbol{\varphi}_{n,1}) - \boldsymbol{\varphi}_{n,1}^\top \boldsymbol{\varphi}_{n,1}) \boldsymbol{X}_n \boldsymbol{W}_V \boldsymbol{w}_O \boldsymbol{\mu}_-^\top \\
&+ \sum_{i=2}^M \boldsymbol{X}_n^\top (diag(\boldsymbol{\varphi}_{n,i}) - \boldsymbol{\varphi}_{n,i}^\top \boldsymbol{\varphi}_{n,i}) \boldsymbol{X}_n \boldsymbol{W}_V \boldsymbol{w}_O \boldsymbol{\xi}_{n,i}^\top) \boldsymbol{W}_Q.
\end{aligned}
\tag{179}
$$

We will give expressions for $\alpha$ and $\beta$ defined in Definition B.3 based on these two equations above. By (178) and the orthogonal relation between $\boldsymbol{\mu}$ and $\boldsymbol{\xi}$, we have

$$
\begin{aligned}
\Delta \boldsymbol{q}_+^{(t)} = \boldsymbol{\mu}_+^\top \Delta \boldsymbol{W}_Q^{(t)} &= \frac{\eta}{NM} \sum_{n \in S_+} -\ell'^{(t)}_n \|\boldsymbol{\mu}\|_2^2 \boldsymbol{w}_O^\top \boldsymbol{W}_V^{(t)\top} \boldsymbol{X}_n^\top (diag(\boldsymbol{\varphi}_{n,1}) - \boldsymbol{\varphi}_{n,1}^\top \boldsymbol{\varphi}_{n,1}) \boldsymbol{X}_n \boldsymbol{W}_K^{(t)} \\
&= \alpha_{+,+}^{(t)} \boldsymbol{k}_+^{(t)} + \sum_{n \in S_+} \sum_{i=2}^M \alpha_{n,+,i}^{(t)} \boldsymbol{k}_{n,i}^{(t)}
\end{aligned}
\tag{180}
$$

where $\boldsymbol{w}_O^\top \boldsymbol{W}_V^{(t)\top} \boldsymbol{X}_n^\top$ and $\boldsymbol{X}_n \boldsymbol{W}_K^{(t)}$ can be viewed in the following forms

$$
\boldsymbol{w}_O^\top \boldsymbol{W}_V^{(t)\top} \boldsymbol{X}_n^\top = \big(V_+^{(t)}, V_{n,2}^{(t)}, \ldots, V_{n,M}^{(t)}\big),
$$

$$
\boldsymbol{X}_n \boldsymbol{W}_K^{(t)} = \big(\boldsymbol{k}_+^{(t)\top}, \boldsymbol{k}_{n,2}^{(t)\top}, \ldots, \boldsymbol{k}_{n,M}^{(t)\top}\big)^\top.
$$

Then we can express $\alpha_{+,+}^{(t)}$ and $\alpha_{n,+,i}^{(t)}$ as follows

$$
\alpha_{+,+}^{(t)} = \frac{\eta}{NM} \sum_{n \in S_+} -\ell_n'^{(t)} \|\boldsymbol{\mu}\|_2^2
$$

$$
\cdot \left( V_+^{(t)} \left( \frac{\exp(\langle \boldsymbol{q}_+^{(t)}, \boldsymbol{k}_+^{(t)} \rangle)}{\exp(\langle \boldsymbol{q}_+^{(t)}, \boldsymbol{k}_+^{(t)} \rangle) + \sum\limits_{j=2}^{M} \exp(\langle \boldsymbol{q}_+^{(t)}, \boldsymbol{k}_{n,j}^{(t)} \rangle)} \right.\right.
$$

$$
\left. - \left( \frac{\exp(\langle \boldsymbol{q}_+^{(t)}, \boldsymbol{k}_+^{(t)} \rangle)}{\exp(\langle \boldsymbol{q}_+^{(t)}, \boldsymbol{k}_+^{(t)} \rangle) + \sum\limits_{j=2}^{M} \exp(\langle \boldsymbol{q}_+^{(t)}, \boldsymbol{k}_{n,j}^{(t)} \rangle)} \right)^2 \right) \tag{181}
$$

$$
- \sum_{i=2}^{M} \left( V_{n,i}^{(t)} \cdot \frac{\exp(\langle \boldsymbol{q}_+^{(t)}, \boldsymbol{k}_+^{(t)} \rangle)}{\exp(\langle \boldsymbol{q}_+^{(t)}, \boldsymbol{k}_+^{(t)} \rangle) + \sum\limits_{j=2}^{M} \exp(\langle \boldsymbol{q}_+^{(t)}, \boldsymbol{k}_{n,j}^{(t)} \rangle)} \right.
$$

$$
\left.\left. \cdot \frac{\exp(\langle \boldsymbol{q}_+^{(t)}, \boldsymbol{k}_{n,i}^{(t)} \rangle)}{\exp(\langle \boldsymbol{q}_+^{(t)}, \boldsymbol{k}_+^{(t)} \rangle) + \sum\limits_{j=2}^{M} \exp(\langle \boldsymbol{q}_+^{(t)}, \boldsymbol{k}_{n,j}^{(t)} \rangle)} \right) \right),
$$

$$
\alpha_{n,+,i}^{(t)} = -\frac{\eta}{NM} \ell_n'^{(t)} \|\boldsymbol{\mu}\|_2^2
$$

$$
\cdot \left( - V_+^{(t)} \cdot \frac{\exp(\langle \boldsymbol{q}_+^{(t)}, \boldsymbol{k}_+^{(t)} \rangle)}{\exp(\langle \boldsymbol{q}_+^{(t)}, \boldsymbol{k}_+^{(t)} \rangle) + \sum\limits_{j=2}^{M} \exp(\langle \boldsymbol{q}_+^{(t)}, \boldsymbol{k}_{n,j}^{(t)} \rangle)} \right.
$$

$$
\cdot \frac{\exp(\langle \boldsymbol{q}_+^{(t)}, \boldsymbol{k}_{n,i}^{(t)} \rangle)}{\exp(\langle \boldsymbol{q}_+^{(t)}, \boldsymbol{k}_+^{(t)} \rangle) + \sum\limits_{j=2}^{M} \exp(\langle \boldsymbol{q}_+^{(t)}, \boldsymbol{k}_{n,j}^{(t)} \rangle)}
$$

$$
+ V_{n,i}^{(t)} \left( \frac{\exp(\langle \boldsymbol{q}_+^{(t)}, \boldsymbol{k}_{n,i}^{(t)} \rangle)}{\exp(\langle \boldsymbol{q}_+^{(t)}, \boldsymbol{k}_+^{(t)} \rangle) + \sum\limits_{j=2}^{M} \exp(\langle \boldsymbol{q}_+^{(t)}, \boldsymbol{k}_{n,j}^{(t)} \rangle)} \right.
$$

$$
\left. - \left( \frac{\exp(\langle \boldsymbol{q}_+^{(t)}, \boldsymbol{k}_{n,i}^{(t)} \rangle)}{\exp(\langle \boldsymbol{q}_+^{(t)}, \boldsymbol{k}_+^{(t)} \rangle) + \sum\limits_{j=2}^{M} \exp(\langle \boldsymbol{q}_+^{(t)}, \boldsymbol{k}_{n,j}^{(t)} \rangle)} \right)^2 \right) \tag{182}
$$

$$
- \sum_{k \neq i} \left( V_{n,k}^{(t)} \cdot \frac{\exp(\langle \boldsymbol{q}_+^{(t)}, \boldsymbol{k}_{n,i}^{(t)} \rangle)}{\exp(\langle \boldsymbol{q}_+^{(t)}, \boldsymbol{k}_+^{(t)} \rangle) + \sum\limits_{j=2}^{M} \exp(\langle \boldsymbol{q}_+^{(t)}, \boldsymbol{k}_{n,j}^{(t)} \rangle)} \right.
$$

$$
\left.\left. \cdot \frac{\exp(\langle \boldsymbol{q}_+^{(t)}, \boldsymbol{k}_{n,k}^{(t)} \rangle)}{\exp(\langle \boldsymbol{q}_+^{(t)}, \boldsymbol{k}_+^{(t)} \rangle) + \sum\limits_{j=2}^{M} \exp(\langle \boldsymbol{q}_+^{(t)}, \boldsymbol{k}_{n,j}^{(t)} \rangle)} \right) \right).
$$

Using the similar method as for $\Delta \boldsymbol{q}_+^{(t)}$, we get the other $\alpha$ and $\beta$ as follows

$$\alpha_{-,-}^{(t)} = \frac{\eta}{NM} \sum_{n \in S_-} \ell_n'^{(t)} \|\boldsymbol{\mu}\|_2^2$$

$$\cdot \Big( V_-^{(t)} \big( \frac{\exp(\langle \boldsymbol{q}_-^{(t)}, \boldsymbol{k}_-^{(t)} \rangle)}{\exp(\langle \boldsymbol{q}_-^{(t)}, \boldsymbol{k}_-^{(t)} \rangle) + \sum\limits_{j=2}^{M} \exp(\langle \boldsymbol{q}_-^{(t)}, \boldsymbol{k}_{n,j}^{(t)} \rangle)}$$

$$- \big( \frac{\exp(\langle \boldsymbol{q}_-^{(t)}, \boldsymbol{k}_-^{(t)} \rangle)}{\exp(\langle \boldsymbol{q}_-^{(t)}, \boldsymbol{k}_-^{(t)} \rangle) + \sum\limits_{j=2}^{M} \exp(\langle \boldsymbol{q}_-^{(t)}, \boldsymbol{k}_{n,j}^{(t)} \rangle)} \big)^2 \big) \tag{183}$$

$$- \sum_{i=2}^{M} \big( V_{n,i}^{(t)} \cdot \frac{\exp(\langle \boldsymbol{q}_-^{(t)}, \boldsymbol{k}_-^{(t)} \rangle)}{\exp(\langle \boldsymbol{q}_-^{(t)}, \boldsymbol{k}_-^{(t)} \rangle) + \sum\limits_{j=2}^{M} \exp(\langle \boldsymbol{q}_-^{(t)}, \boldsymbol{k}_{n,j}^{(t)} \rangle)}$$

$$\cdot \frac{\exp(\langle \boldsymbol{q}_-^{(t)}, \boldsymbol{k}_{n,i}^{(t)} \rangle)}{\exp(\langle \boldsymbol{q}_-^{(t)}, \boldsymbol{k}_-^{(t)} \rangle) + \sum\limits_{j=2}^{M} \exp(\langle \boldsymbol{q}_-^{(t)}, \boldsymbol{k}_{n,j}^{(t)} \rangle)} \big) \Big),$$

$$\alpha_{n,-,i}^{(t)} = \frac{\eta}{NM} \ell_n'^{(t)} \|\boldsymbol{\mu}\|_2^2$$

$$\cdot \Big( - V_-^{(t)} \cdot \frac{\exp(\langle \boldsymbol{q}_-^{(t)}, \boldsymbol{k}_-^{(t)} \rangle)}{\exp(\langle \boldsymbol{q}_-^{(t)}, \boldsymbol{k}_-^{(t)} \rangle) + \sum\limits_{j=2}^{M} \exp(\langle \boldsymbol{q}_-^{(t)}, \boldsymbol{k}_{n,j}^{(t)} \rangle)}$$

$$\cdot \frac{\exp(\langle \boldsymbol{q}_-^{(t)}, \boldsymbol{k}_{n,i}^{(t)} \rangle)}{\exp(\langle \boldsymbol{q}_-^{(t)}, \boldsymbol{k}_-^{(t)} \rangle) + \sum\limits_{j=2}^{M} \exp(\langle \boldsymbol{q}_-^{(t)}, \boldsymbol{k}_{n,j}^{(t)} \rangle)}$$

$$+ V_{n,i}^{(t)} \big( \frac{\exp(\langle \boldsymbol{q}_-^{(t)}, \boldsymbol{k}_{n,i}^{(t)} \rangle)}{\exp(\langle \boldsymbol{q}_-^{(t)}, \boldsymbol{k}_-^{(t)} \rangle) + \sum\limits_{j=2}^{M} \exp(\langle \boldsymbol{q}_-^{(t)}, \boldsymbol{k}_{n,j}^{(t)} \rangle)} \tag{184}$$

$$- \big( \frac{\exp(\langle \boldsymbol{q}_-^{(t)}, \boldsymbol{k}_{n,i}^{(t)} \rangle)}{\exp(\langle \boldsymbol{q}_-^{(t)}, \boldsymbol{k}_-^{(t)} \rangle) + \sum\limits_{j=2}^{M} \exp(\langle \boldsymbol{q}_-^{(t)}, \boldsymbol{k}_{n,j}^{(t)} \rangle)} \big)^2 \big)$$

$$- \sum_{k \neq i} \big( V_{n,k}^{(t)} \cdot \frac{\exp(\langle \boldsymbol{q}_-^{(t)}, \boldsymbol{k}_{n,i}^{(t)} \rangle)}{\exp(\langle \boldsymbol{q}_-^{(t)}, \boldsymbol{k}_-^{(t)} \rangle) + \sum\limits_{j=2}^{M} \exp(\langle \boldsymbol{q}_-^{(t)}, \boldsymbol{k}_{n,j}^{(t)} \rangle)}$$

$$\cdot \frac{\exp(\langle \boldsymbol{q}_-^{(t)}, \boldsymbol{k}_{n,k}^{(t)} \rangle)}{\exp(\langle \boldsymbol{q}_-^{(t)}, \boldsymbol{k}_-^{(t)} \rangle) + \sum\limits_{j=2}^{M} \exp(\langle \boldsymbol{q}_-^{(t)}, \boldsymbol{k}_{n,j}^{(t)} \rangle)} \big) \Big),$$

$$
\alpha_{n',i',+}^{(t)} = \frac{\eta}{NM} \sum_{n \in S_+} -\ell_n'^{(t)} \sum_{i=2}^{M} \langle \boldsymbol{\xi}_{n',i'}, \boldsymbol{\xi}_{n,i} \rangle
$$

$$
\cdot \left( V_+^{(t)} \left( \frac{\exp(\langle \boldsymbol{q}_{n,i}^{(t)}, \boldsymbol{k}_+^{(t)} \rangle)}{\exp(\langle \boldsymbol{q}_{n,i}^{(t)}, \boldsymbol{k}_+^{(t)} \rangle) + \sum\limits_{j=2}^{M} \exp(\langle \boldsymbol{q}_{n,i}^{(t)}, \boldsymbol{k}_{n,j}^{(t)} \rangle)} \right.\right.
$$

$$
- \left( \frac{\exp(\langle \boldsymbol{q}_{n,i}^{(t)}, \boldsymbol{k}_+^{(t)} \rangle)}{\exp(\langle \boldsymbol{q}_{n,i}^{(t)}, \boldsymbol{k}_+^{(t)} \rangle) + \sum\limits_{j=2}^{M} \exp(\langle \boldsymbol{q}_{n,i}^{(t)}, \boldsymbol{k}_{n,j}^{(t)} \rangle)} \right)^2 \right) \tag{185}
$$

$$
- \sum_{k=2}^{M} \left( V_{n,i}^{(t)} \cdot \frac{\exp(\langle \boldsymbol{q}_{n,i}^{(t)}, \boldsymbol{k}_+^{(t)} \rangle)}{\exp(\langle \boldsymbol{q}_{n,i}^{(t)}, \boldsymbol{k}_+^{(t)} \rangle) + \sum\limits_{j=2}^{M} \exp(\langle \boldsymbol{q}_{n,i}^{(t)}, \boldsymbol{k}_{n,j}^{(t)} \rangle)} \right.
$$

$$
\left.\left.\left. \cdot \frac{\exp(\langle \boldsymbol{q}_{n,i}^{(t)}, \boldsymbol{k}_{n,k}^{(t)} \rangle)}{\exp(\langle \boldsymbol{q}_{n,i}^{(t)}, \boldsymbol{k}_+^{(t)} \rangle) + \sum\limits_{j=2}^{M} \exp(\langle \boldsymbol{q}_{n,i}^{(t)}, \boldsymbol{k}_{n,j}^{(t)} \rangle)} \right) \right) \right),
$$

$$
\alpha_{n',i',-}^{(t)} = \frac{\eta}{NM} \sum_{n \in S_-} \ell_n'^{(t)} \sum_{i=2}^{M} \langle \boldsymbol{\xi}_{n',i'}, \boldsymbol{\xi}_{n,i} \rangle
$$

$$
\cdot \left( V_-^{(t)} \left( \frac{\exp(\langle \boldsymbol{q}_{n,i}^{(t)}, \boldsymbol{k}_-^{(t)} \rangle)}{\exp(\langle \boldsymbol{q}_{n,i}^{(t)}, \boldsymbol{k}_-^{(t)} \rangle) + \sum\limits_{j=2}^{M} \exp(\langle \boldsymbol{q}_{n,i}^{(t)}, \boldsymbol{k}_{n,j}^{(t)} \rangle)} \right.\right.
$$

$$
- \left( \frac{\exp(\langle \boldsymbol{q}_{n,i}^{(t)}, \boldsymbol{k}_-^{(t)} \rangle)}{\exp(\langle \boldsymbol{q}_{n,i}^{(t)}, \boldsymbol{k}_-^{(t)} \rangle) + \sum\limits_{j=2}^{M} \exp(\langle \boldsymbol{q}_{n,i}^{(t)}, \boldsymbol{k}_{n,j}^{(t)} \rangle)} \right)^2 \right) \tag{186}
$$

$$
- \sum_{k=2}^{M} \left( V_{n,i}^{(t)} \cdot \frac{\exp(\langle \boldsymbol{q}_{n,i}^{(t)}, \boldsymbol{k}_-^{(t)} \rangle)}{\exp(\langle \boldsymbol{q}_{n,i}^{(t)}, \boldsymbol{k}_-^{(t)} \rangle) + \sum\limits_{j=2}^{M} \exp(\langle \boldsymbol{q}_{n,i}^{(t)}, \boldsymbol{k}_{n,j}^{(t)} \rangle)} \right.
$$

$$
\left.\left.\left. \cdot \frac{\exp(\langle \boldsymbol{q}_{n,i}^{(t)}, \boldsymbol{k}_{n,k}^{(t)} \rangle)}{\exp(\langle \boldsymbol{q}_{n,i}^{(t)}, \boldsymbol{k}_-^{(t)} \rangle) + \sum\limits_{j=2}^{M} \exp(\langle \boldsymbol{q}_{n,i}^{(t)}, \boldsymbol{k}_{n,j}^{(t)} \rangle)} \right) \right) \right),
$$

$$
\alpha_{n',i',n,i}^{(t)} = \frac{\eta}{NM} - \ell_n'^{(t)} \sum_{k=2}^{M} \langle \boldsymbol{\xi}_{n',i'}, \boldsymbol{\xi}_{n,k} \rangle
$$

$$
\cdot \Big( - V_+^{(t)} \cdot \frac{\exp(\langle \boldsymbol{q}_{n,k}^{(t)}, \boldsymbol{k}_+^{(t)} \rangle)}{\exp(\langle \boldsymbol{q}_{n,k}^{(t)}, \boldsymbol{k}_+^{(t)} \rangle) + \sum\limits_{j=2}^{M} \exp(\langle \boldsymbol{q}_{n,k}^{(t)}, \boldsymbol{k}_{n,j}^{(t)} \rangle)}
$$

$$
\cdot \frac{\exp(\langle \boldsymbol{q}_{n,k}^{(t)}, \boldsymbol{k}_{n,i}^{(t)} \rangle)}{\exp(\langle \boldsymbol{q}_{n,k}^{(t)}, \boldsymbol{k}_+^{(t)} \rangle) + \sum\limits_{j=2}^{M} \exp(\langle \boldsymbol{q}_{n,k}^{(t)}, \boldsymbol{k}_{n,j}^{(t)} \rangle)}
$$

$$
+ V_{n,i}^{(t)} \Big( \frac{\exp(\langle \boldsymbol{q}_{n,k}^{(t)}, \boldsymbol{k}_{n,i}^{(t)} \rangle)}{\exp(\langle \boldsymbol{q}_{n,k}^{(t)}, \boldsymbol{k}_+^{(t)} \rangle) + \sum\limits_{j=2}^{M} \exp(\langle \boldsymbol{q}_{n,k}^{(t)}, \boldsymbol{k}_{n,j}^{(t)} \rangle)} \tag{187}
$$

$$
- \Big( \frac{\exp(\langle \boldsymbol{q}_{n,k}^{(t)}, \boldsymbol{k}_{n,i}^{(t)} \rangle)}{\exp(\langle \boldsymbol{q}_{n,k}^{(t)}, \boldsymbol{k}_+^{(t)} \rangle) + \sum\limits_{j=2}^{M} \exp(\langle \boldsymbol{q}_{n,k}^{(t)}, \boldsymbol{k}_{n,j}^{(t)} \rangle)} \Big)^2 \Big)
$$

$$
- \sum_{l \neq i} \Big( V_{n,l}^{(t)} \cdot \frac{\exp(\langle \boldsymbol{q}_{n,k}^{(t)}, \boldsymbol{k}_{n,i}^{(t)} \rangle)}{\exp(\langle \boldsymbol{q}_{n,k}^{(t)}, \boldsymbol{k}_+^{(t)} \rangle) + \sum\limits_{j=2}^{M} \exp(\langle \boldsymbol{q}_{n,k}^{(t)}, \boldsymbol{k}_{n,j}^{(t)} \rangle)}
$$

$$
\cdot \frac{\exp(\langle \boldsymbol{q}_{n,k}^{(t)}, \boldsymbol{k}_{n,l}^{(t)} \rangle)}{\exp(\langle \boldsymbol{q}_{n,k}^{(t)}, \boldsymbol{k}_+^{(t)} \rangle) + \sum\limits_{j=2}^{M} \exp(\langle \boldsymbol{q}_{n,k}^{(t)}, \boldsymbol{k}_{n,j}^{(t)} \rangle)} \Big) \Big)
$$

for $n \in S_+$,

$$\alpha_{n',i',n,i}^{(t)} = \frac{\eta}{NM} \ell_n'^{(t)} \sum_{k=2}^{M} \langle \boldsymbol{\xi}_{n',i'}, \boldsymbol{\xi}_{n,k} \rangle$$

$$\cdot \Bigg( - V_-^{(t)} \cdot \frac{\exp(\langle \boldsymbol{q}_{n,k}^{(t)}, \boldsymbol{k}_-^{(t)} \rangle)}{\exp(\langle \boldsymbol{q}_{n,k}^{(t)}, \boldsymbol{k}_-^{(t)} \rangle) + \sum\limits_{j=2}^{M} \exp(\langle \boldsymbol{q}_{n,k}^{(t)}, \boldsymbol{k}_{n,j}^{(t)} \rangle)}$$

$$\cdot \frac{\exp(\langle \boldsymbol{q}_{n,k}^{(t)}, \boldsymbol{k}_{n,i}^{(t)} \rangle)}{\exp(\langle \boldsymbol{q}_{n,k}^{(t)}, \boldsymbol{k}_-^{(t)} \rangle) + \sum\limits_{j=2}^{M} \exp(\langle \boldsymbol{q}_{n,k}^{(t)}, \boldsymbol{k}_{n,j}^{(t)} \rangle)}$$

$$+ V_{n,i}^{(t)} \Big( \frac{\exp(\langle \boldsymbol{q}_{n,k}^{(t)}, \boldsymbol{k}_{n,i}^{(t)} \rangle)}{\exp(\langle \boldsymbol{q}_{n,k}^{(t)}, \boldsymbol{k}_-^{(t)} \rangle) + \sum\limits_{j=2}^{M} \exp(\langle \boldsymbol{q}_{n,k}^{(t)}, \boldsymbol{k}_{n,j}^{(t)} \rangle)} \tag{188}$$

$$- \Big( \frac{\exp(\langle \boldsymbol{q}_{n,k}^{(t)}, \boldsymbol{k}_{n,i}^{(t)} \rangle)}{\exp(\langle \boldsymbol{q}_{n,k}^{(t)}, \boldsymbol{k}_-^{(t)} \rangle) + \sum\limits_{j=2}^{M} \exp(\langle \boldsymbol{q}_{n,k}^{(t)}, \boldsymbol{k}_{n,j}^{(t)} \rangle)} \Big)^2 \Big)$$

$$- \sum_{l \neq i} \Big( V_{n,l}^{(t)} \cdot \frac{\exp(\langle \boldsymbol{q}_{n,k}^{(t)}, \boldsymbol{k}_{n,i}^{(t)} \rangle)}{\exp(\langle \boldsymbol{q}_{n,k}^{(t)}, \boldsymbol{k}_-^{(t)} \rangle) + \sum\limits_{j=2}^{M} \exp(\langle \boldsymbol{q}_{n,k}^{(t)}, \boldsymbol{k}_{n,j}^{(t)} \rangle)}$$

$$\cdot \frac{\exp(\langle \boldsymbol{q}_{n,k}^{(t)}, \boldsymbol{k}_{n,l}^{(t)} \rangle)}{\exp(\langle \boldsymbol{q}_{n,k}^{(t)}, \boldsymbol{k}_-^{(t)} \rangle) + \sum\limits_{j=2}^{M} \exp(\langle \boldsymbol{q}_{n,k}^{(t)}, \boldsymbol{k}_{n,j}^{(t)} \rangle)} \Big) \Bigg)$$

for $n \in S_-$,

$$\beta_{-,-}^{(t)} = \frac{\eta \|\boldsymbol{\mu}\|_2^2}{NM} \sum_{n \in S_-} \ell_n'^{(t)}$$

$$\cdot \Bigg( V_-^{(t)} \Big( \frac{\exp(\langle \boldsymbol{q}_-^{(t)}, \boldsymbol{k}_-^{(t)} \rangle)}{\exp(\langle \boldsymbol{q}_-^{(t)}, \boldsymbol{k}_-^{(t)} \rangle) + \sum\limits_{j=2}^{M} \exp(\langle \boldsymbol{q}_-^{(t)}, \boldsymbol{k}_{n,j}^{(t)} \rangle)}$$

$$- \Big( \frac{\exp(\langle \boldsymbol{q}_-^{(t)}, \boldsymbol{k}_-^{(t)} \rangle)}{\exp(\langle \boldsymbol{q}_-^{(t)}, \boldsymbol{k}_-^{(t)} \rangle) + \sum\limits_{j=2}^{M} \exp(\langle \boldsymbol{q}_-^{(t)}, \boldsymbol{k}_{n,j}^{(t)} \rangle)} \Big)^2 \Big) \tag{189}$$

$$- \sum_{i=2}^{M} \Big( V_{n,i}^{(t)} \cdot \frac{\exp(\langle \boldsymbol{q}_-^{(t)}, \boldsymbol{k}_-^{(t)} \rangle)}{\exp(\langle \boldsymbol{q}_-^{(t)}, \boldsymbol{k}_-^{(t)} \rangle) + \sum\limits_{j=2}^{M} \exp(\langle \boldsymbol{q}_-^{(t)}, \boldsymbol{k}_{n,j}^{(t)} \rangle)}$$

$$\cdot \frac{\exp(\langle \boldsymbol{q}_-^{(t)}, \boldsymbol{k}_{n,i}^{(t)} \rangle)}{\exp(\langle \boldsymbol{q}_-^{(t)}, \boldsymbol{k}_-^{(t)} \rangle) + \sum\limits_{j=2}^{M} \exp(\langle \boldsymbol{q}_-^{(t)}, \boldsymbol{k}_{n,j}^{(t)} \rangle)} \Big) \Bigg),$$

$$
\beta_{n,+,i}^{(t)} = -\frac{\eta \|\boldsymbol{\mu}\|_2^2}{NM} \ell_n'^{(t)}
$$
$$
\cdot \Big( V_+^{(t)} \big( \frac{\exp(\langle \boldsymbol{q}_{n,i}^{(t)}, \boldsymbol{k}_+^{(t)} \rangle)}{\exp(\langle \boldsymbol{q}_{n,i}^{(t)}, \boldsymbol{k}_+^{(t)} \rangle) + \sum\limits_{j=2}^{M} \exp(\langle \boldsymbol{q}_{n,i}^{(t)}, \boldsymbol{k}_{n,j}^{(t)} \rangle)}
$$
$$
- \big( \frac{\exp(\langle \boldsymbol{q}_{n,i}^{(t)}, \boldsymbol{k}_+^{(t)} \rangle)}{\exp(\langle \boldsymbol{q}_{n,i}^{(t)}, \boldsymbol{k}_+^{(t)} \rangle) + \sum\limits_{j=2}^{M} \exp(\langle \boldsymbol{q}_{n,i}^{(t)}, \boldsymbol{k}_{n,j}^{(t)} \rangle)} \big)^2 \big) \tag{190}
$$
$$
- \sum_{k=2}^{M} \big( V_{n,i}^{(t)} \cdot \frac{\exp(\langle \boldsymbol{q}_{n,i}^{(t)}, \boldsymbol{k}_+^{(t)} \rangle)}{\exp(\langle \boldsymbol{q}_{n,i}^{(t)}, \boldsymbol{k}_+^{(t)} \rangle) + \sum\limits_{j=2}^{M} \exp(\langle \boldsymbol{q}_{n,i}^{(t)}, \boldsymbol{k}_{n,j}^{(t)} \rangle)}
$$
$$
\cdot \frac{\exp(\langle \boldsymbol{q}_{n,i}^{(t)}, \boldsymbol{k}_{n,k}^{(t)} \rangle)}{\exp(\langle \boldsymbol{q}_{n,i}^{(t)}, \boldsymbol{k}_+^{(t)} \rangle) + \sum\limits_{j=2}^{M} \exp(\langle \boldsymbol{q}_{n,i}^{(t)}, \boldsymbol{k}_{n,j}^{(t)} \rangle)} \big) \Big),
$$

$$
\beta_{n,-,i}^{(t)} = \frac{\eta \|\boldsymbol{\mu}\|_2^2}{NM} \ell_n'^{(t)}
$$
$$
\cdot \Big( V_-^{(t)} \big( \frac{\exp(\langle \boldsymbol{q}_{n,i}^{(t)}, \boldsymbol{k}_-^{(t)} \rangle)}{\exp(\langle \boldsymbol{q}_{n,i}^{(t)}, \boldsymbol{k}_-^{(t)} \rangle) + \sum\limits_{j=2}^{M} \exp(\langle \boldsymbol{q}_{n,i}^{(t)}, \boldsymbol{k}_{n,j}^{(t)} \rangle)}
$$
$$
- \big( \frac{\exp(\langle \boldsymbol{q}_{n,i}^{(t)}, \boldsymbol{k}_-^{(t)} \rangle)}{\exp(\langle \boldsymbol{q}_{n,i}^{(t)}, \boldsymbol{k}_-^{(t)} \rangle) + \sum\limits_{j=2}^{M} \exp(\langle \boldsymbol{q}_{n,i}^{(t)}, \boldsymbol{k}_{n,j}^{(t)} \rangle)} \big)^2 \big) \tag{191}
$$
$$
- \sum_{k=2}^{M} \big( V_{n,i}^{(t)} \cdot \frac{\exp(\langle \boldsymbol{q}_{n,i}^{(t)}, \boldsymbol{k}_-^{(t)} \rangle)}{\exp(\langle \boldsymbol{q}_{n,i}^{(t)}, \boldsymbol{k}_-^{(t)} \rangle) + \sum\limits_{j=2}^{M} \exp(\langle \boldsymbol{q}_{n,i}^{(t)}, \boldsymbol{k}_{n,j}^{(t)} \rangle)}
$$
$$
\cdot \frac{\exp(\langle \boldsymbol{q}_{n,i}^{(t)}, \boldsymbol{k}_{n,k}^{(t)} \rangle)}{\exp(\langle \boldsymbol{q}_{n,i}^{(t)}, \boldsymbol{k}_-^{(t)} \rangle) + \sum\limits_{j=2}^{M} \exp(\langle \boldsymbol{q}_{n,i}^{(t)}, \boldsymbol{k}_{n,j}^{(t)} \rangle)} \big) \Big),
$$

$$\beta_{n',i',+}^{(t)} = \frac{\eta}{NM} \sum_{n \in S_+} -\ell_n'^{(t)}$$

$$\left( -V_+^{(t)} \sum_{i=2}^{M} \left( \langle \boldsymbol{\xi}_{n',i'}, \boldsymbol{\xi}_{n,i} \rangle \frac{\exp(\langle \boldsymbol{q}_+^{(t)}, \boldsymbol{k}_+^{(t)} \rangle)}{\exp(\langle \boldsymbol{q}_+^{(t)}, \boldsymbol{k}_+^{(t)} \rangle) + \sum\limits_{j=2}^{M} \exp(\langle \boldsymbol{q}_+^{(t)}, \boldsymbol{k}_{n,j}^{(t)} \rangle)} \right. \right.$$

$$\cdot \frac{\exp(\langle \boldsymbol{q}_+^{(t)}, \boldsymbol{k}_{n,i}^{(t)} \rangle)}{\exp(\langle \boldsymbol{q}_+^{(t)}, \boldsymbol{k}_+^{(t)} \rangle) + \sum\limits_{j=2}^{M} \exp(\langle \boldsymbol{q}_+^{(t)}, \boldsymbol{k}_{n,j}^{(t)} \rangle)} \Big)$$

$$+ \sum_{k=2}^{M} V_{n,k}^{(t)} \Big( \langle \boldsymbol{\xi}_{n',i'}, \boldsymbol{\xi}_{n,k} \rangle \frac{\exp(\langle \boldsymbol{q}_+^{(t)}, \boldsymbol{k}_{n,k}^{(t)} \rangle)}{\exp(\langle \boldsymbol{q}_+^{(t)}, \boldsymbol{k}_+^{(t)} \rangle) + \sum\limits_{j=2}^{M} \exp(\langle \boldsymbol{q}_+^{(t)}, \boldsymbol{k}_{n,j}^{(t)} \rangle)} \tag{192}$$

$$- \sum_{i=2}^{M} \Big( \langle \boldsymbol{\xi}_{n',i'}, \boldsymbol{\xi}_{n,i} \rangle \frac{\exp(\langle \boldsymbol{q}_+^{(t)}, \boldsymbol{k}_{n,k}^{(t)} \rangle)}{\exp(\langle \boldsymbol{q}_+^{(t)}, \boldsymbol{k}_+^{(t)} \rangle) + \sum\limits_{j=2}^{M} \exp(\langle \boldsymbol{q}_+^{(t)}, \boldsymbol{k}_{n,j}^{(t)} \rangle)}$$

$$\left. \left. \cdot \frac{\exp(\langle \boldsymbol{q}_+^{(t)}, \boldsymbol{k}_{n,i}^{(t)} \rangle)}{\exp(\langle \boldsymbol{q}_+^{(t)}, \boldsymbol{k}_+^{(t)} \rangle) + \sum\limits_{j=2}^{M} \exp(\langle \boldsymbol{q}_+^{(t)}, \boldsymbol{k}_{n,j}^{(t)} \rangle)} \Big) \Big) \right) \right),$$

$$\beta_{n',i',-}^{(t)} = \frac{\eta}{NM} \sum_{n \in S_-} \ell_n'^{(t)}$$

$$\left( -V_-^{(t)} \sum_{i=2}^{M} \left( \langle \boldsymbol{\xi}_{n',i'}, \boldsymbol{\xi}_{n,i} \rangle \frac{\exp(\langle \boldsymbol{q}_-^{(t)}, \boldsymbol{k}_-^{(t)} \rangle)}{\exp(\langle \boldsymbol{q}_-^{(t)}, \boldsymbol{k}_-^{(t)} \rangle) + \sum\limits_{j=2}^{M} \exp(\langle \boldsymbol{q}_-^{(t)}, \boldsymbol{k}_{n,j}^{(t)} \rangle)} \right. \right.$$

$$\cdot \frac{\exp(\langle \boldsymbol{q}_-^{(t)}, \boldsymbol{k}_{n,i}^{(t)} \rangle)}{\exp(\langle \boldsymbol{q}_-^{(t)}, \boldsymbol{k}_-^{(t)} \rangle) + \sum\limits_{j=2}^{M} \exp(\langle \boldsymbol{q}_-^{(t)}, \boldsymbol{k}_{n,j}^{(t)} \rangle)} \Big)$$

$$+ \sum_{k=2}^{M} V_{n,k}^{(t)} \Big( \langle \boldsymbol{\xi}_{n',i'}, \boldsymbol{\xi}_{n,k} \rangle \frac{\exp(\langle \boldsymbol{q}_-^{(t)}, \boldsymbol{k}_{n,k}^{(t)} \rangle)}{\exp(\langle \boldsymbol{q}_-^{(t)}, \boldsymbol{k}_-^{(t)} \rangle) + \sum\limits_{j=2}^{M} \exp(\langle \boldsymbol{q}_-^{(t)}, \boldsymbol{k}_{n,j}^{(t)} \rangle)} \tag{193}$$

$$- \sum_{i=2}^{M} \Big( \langle \boldsymbol{\xi}_{n',i'}, \boldsymbol{\xi}_{n,i} \rangle \frac{\exp(\langle \boldsymbol{q}_-^{(t)}, \boldsymbol{k}_{n,k}^{(t)} \rangle)}{\exp(\langle \boldsymbol{q}_-^{(t)}, \boldsymbol{k}_-^{(t)} \rangle) + \sum\limits_{j=2}^{M} \exp(\langle \boldsymbol{q}_-^{(t)}, \boldsymbol{k}_{n,j}^{(t)} \rangle)}$$

$$\left. \left. \cdot \frac{\exp(\langle \boldsymbol{q}_-^{(t)}, \boldsymbol{k}_{n,i}^{(t)} \rangle)}{\exp(\langle \boldsymbol{q}_-^{(t)}, \boldsymbol{k}_-^{(t)} \rangle) + \sum\limits_{j=2}^{M} \exp(\langle \boldsymbol{q}_-^{(t)}, \boldsymbol{k}_{n,j}^{(t)} \rangle)} \Big) \Big) \right) \right),$$

$$\beta_{n',i',n,i}^{(t)} = \frac{\eta}{NM} - \ell_n'^{(t)}$$

$$\left( - V_+^{(t)} \sum_{k=2}^{M} \left( \langle \boldsymbol{\xi}_{n',i'}, \boldsymbol{\xi}_{n,k} \rangle \frac{\exp(\langle \boldsymbol{q}_{n,i}^{(t)}, \boldsymbol{k}_+^{(t)} \rangle)}{\exp(\langle \boldsymbol{q}_{n,i}^{(t)}, \boldsymbol{k}_+^{(t)} \rangle) + \sum_{j=2}^{M} \exp(\langle \boldsymbol{q}_{n,i}^{(t)}, \boldsymbol{k}_{n,j}^{(t)} \rangle)} \right.$$

$$\cdot \frac{\exp(\langle \boldsymbol{q}_{n,i}^{(t)}, \boldsymbol{k}_{n,k}^{(t)} \rangle)}{\exp(\langle \boldsymbol{q}_{n,i}^{(t)}, \boldsymbol{k}_+^{(t)} \rangle) + \sum_{j=2}^{M} \exp(\langle \boldsymbol{q}_{n,i}^{(t)}, \boldsymbol{k}_{n,j}^{(t)} \rangle)} )$$

$$+ \sum_{k=2}^{M} V_{n,k}^{(t)} \left( \langle \boldsymbol{\xi}_{n',i'}, \boldsymbol{\xi}_{n,k} \rangle \frac{\exp(\langle \boldsymbol{q}_{n,i}^{(t)}, \boldsymbol{k}_{n,k}^{(t)} \rangle)}{\exp(\langle \boldsymbol{q}_{n,i}^{(t)}, \boldsymbol{k}_+^{(t)} \rangle) + \sum_{j=2}^{M} \exp(\langle \boldsymbol{q}_{n,i}^{(t)}, \boldsymbol{k}_{n,j}^{(t)} \rangle)} \right. \tag{194}$$

$$- \sum_{l=2}^{M} \left( \langle \boldsymbol{\xi}_{n',i'}, \boldsymbol{\xi}_{n,l} \rangle \frac{\exp(\langle \boldsymbol{q}_{n,i}^{(t)}, \boldsymbol{k}_{n,k}^{(t)} \rangle)}{\exp(\langle \boldsymbol{q}_{n,i}^{(t)}, \boldsymbol{k}_+^{(t)} \rangle) + \sum_{j=2}^{M} \exp(\langle \boldsymbol{q}_{n,i}^{(t)}, \boldsymbol{k}_{n,j}^{(t)} \rangle)} \right.$$

$$\left. \left. \left. \cdot \frac{\exp(\langle \boldsymbol{q}_{n,i}^{(t)}, \boldsymbol{k}_{n,l}^{(t)} \rangle)}{\exp(\langle \boldsymbol{q}_{n,i}^{(t)}, \boldsymbol{k}_+^{(t)} \rangle) + \sum_{j=2}^{M} \exp(\langle \boldsymbol{q}_{n,i}^{(t)}, \boldsymbol{k}_{n,j}^{(t)} \rangle)} \right) \right) \right)$$

for $n \in S_+$,

$$\beta_{n',i',n,i}^{(t)} = \frac{\eta}{NM} \ell_n'^{(t)}$$

$$\left( - V_-^{(t)} \sum_{k=2}^{M} \left( \langle \boldsymbol{\xi}_{n',i'}, \boldsymbol{\xi}_{n,k} \rangle \frac{\exp(\langle \boldsymbol{q}_{n,i}^{(t)}, \boldsymbol{k}_-^{(t)} \rangle)}{\exp(\langle \boldsymbol{q}_{n,i}^{(t)}, \boldsymbol{k}_-^{(t)} \rangle) + \sum_{j=2}^{M} \exp(\langle \boldsymbol{q}_{n,i}^{(t)}, \boldsymbol{k}_{n,j}^{(t)} \rangle)} \right.$$

$$\cdot \frac{\exp(\langle \boldsymbol{q}_{n,i}^{(t)}, \boldsymbol{k}_{n,k}^{(t)} \rangle)}{\exp(\langle \boldsymbol{q}_{n,i}^{(t)}, \boldsymbol{k}_-^{(t)} \rangle) + \sum_{j=2}^{M} \exp(\langle \boldsymbol{q}_{n,i}^{(t)}, \boldsymbol{k}_{n,j}^{(t)} \rangle)} )$$

$$+ \sum_{k=2}^{M} V_{n,k}^{(t)} \left( \langle \boldsymbol{\xi}_{n',i'}, \boldsymbol{\xi}_{n,k} \rangle \frac{\exp(\langle \boldsymbol{q}_{n,i}^{(t)}, \boldsymbol{k}_{n,k}^{(t)} \rangle)}{\exp(\langle \boldsymbol{q}_{n,i}^{(t)}, \boldsymbol{k}_-^{(t)} \rangle) + \sum_{j=2}^{M} \exp(\langle \boldsymbol{q}_{n,i}^{(t)}, \boldsymbol{k}_{n,j}^{(t)} \rangle)} \right. \tag{195}$$

$$- \sum_{l=2}^{M} \left( \langle \boldsymbol{\xi}_{n',i'}, \boldsymbol{\xi}_{n,l} \rangle \frac{\exp(\langle \boldsymbol{q}_{n,i}^{(t)}, \boldsymbol{k}_{n,k}^{(t)} \rangle)}{\exp(\langle \boldsymbol{q}_{n,i}^{(t)}, \boldsymbol{k}_-^{(t)} \rangle) + \sum_{j=2}^{M} \exp(\langle \boldsymbol{q}_{n,i}^{(t)}, \boldsymbol{k}_{n,j}^{(t)} \rangle)} \right.$$

$$\left. \left. \left. \cdot \frac{\exp(\langle \boldsymbol{q}_{n,i}^{(t)}, \boldsymbol{k}_{n,l}^{(t)} \rangle)}{\exp(\langle \boldsymbol{q}_{n,i}^{(t)}, \boldsymbol{k}_-^{(t)} \rangle) + \sum_{j=2}^{M} \exp(\langle \boldsymbol{q}_{n,i}^{(t)}, \boldsymbol{k}_{n,j}^{(t)} \rangle)} \right) \right) \right)$$

for $n \in S_-$.

### F.2 Update Rules for Inner Products

In this subsection, we give the update rules for the inner products of $\boldsymbol{q}$ and $\boldsymbol{k}$.

**Restatement of Lemma B.3.** The dynamics of $\boldsymbol{x} \boldsymbol{W}_Q \boldsymbol{W}_K \boldsymbol{X}^\top$ can be characterized as follows

$$\langle \boldsymbol{q}_+^{(t+1)}, \boldsymbol{k}_+^{(t+1)} \rangle - \langle \boldsymbol{q}_+^{(t)}, \boldsymbol{k}_+^{(t)} \rangle$$

$$= \alpha_{+,+}^{(t)} \|\boldsymbol{k}_+^{(t)}\|_2^2 + \sum_{n \in S_+} \sum_{i=2}^{M} \alpha_{n,+,i}^{(t)} \langle \boldsymbol{k}_+^{(t)}, \boldsymbol{k}_{n,i}^{(t)} \rangle$$

$$+ \beta_{+,+}^{(t)} \|\boldsymbol{q}_+^{(t)}\|_2^2 + \sum_{n \in S_+} \sum_{i=2}^{M} \beta_{n,+,i}^{(t)} \langle \boldsymbol{q}_+^{(t)}, \boldsymbol{q}_{n,i}^{(t)} \rangle$$

$$+ \left( \alpha_{+,+}^{(t)} \boldsymbol{k}_+^{(t)} + \sum_{n \in S_+} \sum_{i=2}^{M} \alpha_{n,+,i}^{(t)} \boldsymbol{k}_{n,i}^{(t)} \right)$$

$$\cdot \left( \beta_{+,+}^{(t)} \boldsymbol{q}_+^{(t)\top} + \sum_{n \in S_+} \sum_{i=2}^{M} \beta_{n,+,i}^{(t)} \boldsymbol{q}_{n,i}^{(t)\top} \right),$$

$$\langle \boldsymbol{q}_-^{(t+1)}, \boldsymbol{k}_-^{(t+1)} \rangle - \langle \boldsymbol{q}_-^{(t)}, \boldsymbol{k}_-^{(t)} \rangle$$

$$= \alpha_{-,-}^{(t)} \|\boldsymbol{k}_-^{(t)}\|_2^2 + \sum_{n \in S_-} \sum_{i=2}^{M} \alpha_{n,-,i}^{(t)} \langle \boldsymbol{k}_-^{(t)}, \boldsymbol{k}_{n,i}^{(t)} \rangle$$

$$+ \beta_{-,-}^{(t)} \|\boldsymbol{q}_-^{(t)}\|_2^2 + \sum_{n \in S_-} \sum_{i=2}^{M} \beta_{n,-,i}^{(t)} \langle \boldsymbol{q}_+^{(t)}, \boldsymbol{q}_{n,i}^{(t)} \rangle$$

$$+ \left( \alpha_{-,-}^{(t)} \boldsymbol{k}_-^{(t)} + \sum_{n \in S_-} \sum_{i=2}^{M} \alpha_{n,-,i}^{(t)} \boldsymbol{k}_{n,i}^{(t)} \right)$$

$$\cdot \left( \beta_{-,-}^{(t)} \boldsymbol{q}_-^{(t)\top} + \sum_{n \in S_-} \sum_{i=2}^{M} \beta_{n,-,i}^{(t)} \boldsymbol{q}_{n,i}^{(t)\top} \right),$$

$$\langle \boldsymbol{q}_{n,i}^{(t+1)}, \boldsymbol{k}_+^{(t+1)} \rangle - \langle \boldsymbol{q}_{n,i}^{(t)}, \boldsymbol{k}_+^{(t)} \rangle$$

$$= \alpha_{n,i,+}^{(t)} \|\boldsymbol{k}_+^{(t)}\|_2^2 + \alpha_{n,i,-}^{(t)} \langle \boldsymbol{k}_+^{(t)}, \boldsymbol{k}_-^{(t)} \rangle + \sum_{n'=1}^{N} \sum_{l=2}^{M} \alpha_{n,i,n',l}^{(t)} \langle \boldsymbol{k}_+^{(t)}, \boldsymbol{k}_{n',l}^{(t)} \rangle$$

$$+ \beta_{+,+}^{(t)} \langle \boldsymbol{q}_+^{(t)}, \boldsymbol{q}_{n,i}^{(t)} \rangle + \sum_{n' \in S_+} \sum_{l=2}^{M} \beta_{n',+,l}^{(t)} \langle \boldsymbol{q}_{n,i}^{(t)}, \boldsymbol{q}_{n',l}^{(t)} \rangle$$

$$+ \left( \alpha_{n,i,+}^{(t)} \boldsymbol{k}_+^{(t)} + \alpha_{n,i,-}^{(t)} \boldsymbol{k}_-^{(t)} + \sum_{n'=1}^{N} \sum_{l=2}^{M} \alpha_{n,i,n',l}^{(t)} \boldsymbol{k}_{n',l}^{(t)} \right)$$

$$\cdot \left( \beta_{+,+}^{(t)} \boldsymbol{q}_+^{(t)\top} + \sum_{n' \in S_+} \sum_{l=2}^{M} \beta_{n',+,l}^{(t)} \boldsymbol{q}_{n',l}^{(t)\top} \right),$$

$$\langle \boldsymbol{q}_{n,i}^{(t+1)}, \boldsymbol{k}_-^{(t+1)} \rangle - \langle \boldsymbol{q}_{n,i}^{(t)}, \boldsymbol{k}_-^{(t)} \rangle$$

$$= \alpha_{n,i,-}^{(t)} \|\boldsymbol{k}_-^{(t)}\|_2^2 + \alpha_{n,i,+}^{(t)} \langle \boldsymbol{k}_+^{(t)}, \boldsymbol{k}_-^{(t)} \rangle + \sum_{n'=1}^{N} \sum_{l=2}^{M} \alpha_{n,i,n',l}^{(t)} \langle \boldsymbol{k}_-^{(t)}, \boldsymbol{k}_{n',l}^{(t)} \rangle$$

$$+ \beta_{-,-}^{(t)} \langle \boldsymbol{q}_-^{(t)}, \boldsymbol{q}_{n,i}^{(t)} \rangle + \sum_{n' \in S_-} \sum_{l=2}^{M} \beta_{n',-,l}^{(t)} \langle \boldsymbol{q}_{n,i}^{(t)}, \boldsymbol{q}_{n',l}^{(t)} \rangle$$

$$+ \left( \alpha_{n,i,+}^{(t)} \boldsymbol{k}_+^{(t)} + \alpha_{n,i,-}^{(t)} \boldsymbol{k}_-^{(t)} + \sum_{n'=1}^{N} \sum_{l=2}^{M} \alpha_{n,i,n',l}^{(t)} \boldsymbol{k}_{n',l}^{(t)} \right)$$

$$\cdot \left( \beta_{-,-}^{(t)} \boldsymbol{q}_-^{(t)\top} + \sum_{n' \in S_-} \sum_{l=2}^{M} \beta_{n',-,l}^{(t)} \boldsymbol{q}_{n',l}^{(t)\top} \right),$$

$$\langle \boldsymbol{q}_+^{(t+1)}, \boldsymbol{k}_{n,j}^{(t+1)} \rangle - \langle \boldsymbol{q}_+^{(t)}, \boldsymbol{k}_{n,j}^{(t)} \rangle$$

$$= \alpha_{+,+}^{(t)} \langle \boldsymbol{k}_+^{(t)}, \boldsymbol{k}_{n,j}^{(t)} \rangle + \sum_{n' \in S_+} \sum_{l=2}^{M} \alpha_{n',+,l}^{(t)} \langle \boldsymbol{k}_{n,j}^{(t)}, \boldsymbol{k}_{n',l}^{(t)} \rangle$$

$$+ \beta_{n,j,+}^{(t)} \|\boldsymbol{q}_+^{(t)}\|_2^2 + \beta_{n,j,-}^{(t)} \langle \boldsymbol{q}_+^{(t)}, \boldsymbol{q}_-^{(t)} \rangle + \sum_{n'=1}^{N} \sum_{l=2}^{M} \beta_{n,j,n',l}^{(t)} \langle \boldsymbol{q}_+^{(t)}, \boldsymbol{q}_{n',l}^{(t)} \rangle$$

$$+ \left( \alpha_{+,+}^{(t)} \boldsymbol{k}_+^{(t)} + \sum_{n' \in S_+} \sum_{l=2}^{M} \alpha_{n',+,l}^{(t)} \boldsymbol{k}_{n',l}^{(t)} \right)$$

$$\cdot \left( \beta_{n,j,+}^{(t)} \boldsymbol{q}_+^{(t)\top} + \beta_{n,j,-}^{(t)} \boldsymbol{q}_-^{(t)\top} + \sum_{n'=1}^{N} \sum_{l=2}^{M} \beta_{n,j,n',l}^{(t)} \boldsymbol{q}_{n',l}^{(t)\top} \right),$$

$$\langle \boldsymbol{q}_-^{(t+1)}, \boldsymbol{k}_{n,j}^{(t+1)} \rangle - \langle \boldsymbol{q}_-^{(t)}, \boldsymbol{k}_{n,j}^{(t)} \rangle$$

$$= \alpha_{-,-}^{(t)} \langle \boldsymbol{k}_-^{(t)}, \boldsymbol{k}_{n,j}^{(t)} \rangle + \sum_{n' \in S_-} \sum_{l=2}^{M} \alpha_{n',-,l}^{(t)} \langle \boldsymbol{k}_{n,j}^{(t)}, \boldsymbol{k}_{n',l}^{(t)} \rangle$$

$$+ \beta_{n,j,-}^{(t)} \|\boldsymbol{q}_-^{(t)}\|_2^2 + \beta_{n,j,+}^{(t)} \langle \boldsymbol{q}_+^{(t)}, \boldsymbol{q}_-^{(t)} \rangle + \sum_{n'=1}^{N} \sum_{l=2}^{M} \beta_{n,j,n',l}^{(t)} \langle \boldsymbol{q}_-^{(t)}, \boldsymbol{q}_{n',l}^{(t)} \rangle$$

$$+ \left( \alpha_{-,-}^{(t)} \boldsymbol{k}_-^{(t)} + \sum_{n' \in S_-} \sum_{l=2}^{M} \alpha_{n',-,l}^{(t)} \boldsymbol{k}_{n',l}^{(t)} \right)$$

$$\cdot \left( \beta_{n,j,+}^{(t)} \boldsymbol{q}_+^{(t)\top} + \beta_{n,j,-}^{(t)} \boldsymbol{q}_-^{(t)\top} + \sum_{n'=1}^{N} \sum_{l=2}^{M} \beta_{n,j,n',l}^{(t)} \boldsymbol{q}_{n',l}^{(t)\top} \right),$$

$$\langle \boldsymbol{q}_{n,i}^{(t+1)}, \boldsymbol{k}_{n,j}^{(t+1)}\rangle - \langle \boldsymbol{q}_{n,i}^{(t)}, \boldsymbol{k}_{n,j}^{(t)}\rangle$$

$$= \alpha_{n,i,+}^{(t)}\langle \boldsymbol{k}_+^{(t)}, \boldsymbol{k}_{n,j}^{(t)}\rangle + \alpha_{n,i,-}^{(t)}\langle \boldsymbol{k}_-^{(t)}, \boldsymbol{k}_{n,j}^{(t)}\rangle + \sum_{n'=1}^{N}\sum_{l=2}^{M}\alpha_{n,i,n',l}^{(t)}\langle \boldsymbol{k}_{n',l}^{(t)}, \boldsymbol{k}_{n,j}^{(t)}\rangle$$

$$+ \beta_{n,j,+}^{(t)}\langle \boldsymbol{q}_+^{(t)}, \boldsymbol{q}_{n,i}^{(t)}\rangle + \beta_{n,j,-}^{(t)}\langle \boldsymbol{q}_-^{(t)}, \boldsymbol{q}_{n,i}^{(t)}\rangle + \sum_{n'=1}^{N}\sum_{l=2}^{M}\beta_{n,j,n',l}^{(t)}\langle \boldsymbol{q}_{n',l}^{(t)}, \boldsymbol{q}_{n,i}^{(t)}\rangle$$

$$+ \left(\alpha_{n,i,+}^{(t)}\boldsymbol{k}_+^{(t)} + \alpha_{n,i,-}^{(t)}\boldsymbol{k}_-^{(t)} + \sum_{n'=1}^{N}\sum_{l=2}^{M}\alpha_{n,i,n',l}^{(t)}\boldsymbol{k}_{n',l}^{(t)}\right)$$

$$\cdot \left(\beta_{n,j,+}^{(t)}\boldsymbol{q}_+^{(t)\top} + \beta_{n,j,-}^{(t)}\boldsymbol{q}_-^{(t)\top} + \sum_{n'=1}^{N}\sum_{l=2}^{M}\beta_{n,j,n',l}^{(t)}\boldsymbol{q}_{n',l}^{(t)\top}\right),$$

for $i, j \in [M]\backslash\{1\}, n \in [N]$.

In addition to these equations above, we give the complete inner product update rule as follows

$$\langle \boldsymbol{q}_{n,i}^{(t+1)}, \boldsymbol{k}_{\overline{n},j}^{(t+1)}\rangle - \langle \boldsymbol{q}_{n,i}^{(t)}, \boldsymbol{k}_{\overline{n},j}^{(t)}\rangle$$

$$= \boldsymbol{\xi}_{n,j}^{\top}(\boldsymbol{W}_Q^{(t)} + \Delta\boldsymbol{W}_Q^{(t)})(\boldsymbol{W}_K^{(t)\top} + \Delta\boldsymbol{W}_K^{(t)\top})\boldsymbol{\xi}_{\overline{n},k} - \langle \boldsymbol{q}_{n,i}^{(t)}, \boldsymbol{k}_{\overline{n},j}^{(t)}\rangle$$

$$= \langle \Delta\boldsymbol{q}_{n,i}^{(t)}, \boldsymbol{k}_{\overline{n},j}^{(t)}\rangle + \langle \boldsymbol{q}_{n,i}^{(t)}, \Delta\boldsymbol{k}_{\overline{n},j}^{(t)}\rangle + \langle \Delta\boldsymbol{q}_{n,i}^{(t)}, \Delta\boldsymbol{k}_{\overline{n},j}^{(t)}\rangle$$

$$= \alpha_{n,i,+}^{(t)}\langle \boldsymbol{k}_+^{(t)}, \boldsymbol{k}_{\overline{n},j}^{(t)}\rangle + \alpha_{n,i,-}^{(t)}\langle \boldsymbol{k}_-^{(t)}, \boldsymbol{k}_{\overline{n},j}^{(t)}\rangle + \sum_{n'=1}^{N}\sum_{l=2}^{M}\alpha_{n,i,n',l}^{(t)}\langle \boldsymbol{k}_{n',l}^{(t)}, \boldsymbol{k}_{\overline{n},j}^{(t)}\rangle$$

$$+ \beta_{\overline{n},j,+}^{(t)}\langle \boldsymbol{q}_+^{(t)}, \boldsymbol{q}_{n,i}^{(t)}\rangle + \beta_{\overline{n},j,-}^{(t)}\langle \boldsymbol{q}_-^{(t)}, \boldsymbol{q}_{n,i}^{(t)}\rangle + \sum_{n'=1}^{N}\sum_{l=2}^{M}\beta_{\overline{n},j,n',l}^{(t)}\langle \boldsymbol{q}_{n',l}^{(t)}, \boldsymbol{q}_{n,i}^{(t)}\rangle \qquad (196)$$

$$+ \left(\alpha_{n,i,+}^{(t)}\boldsymbol{k}_+^{(t)} + \alpha_{n,i,-}^{(t)}\boldsymbol{k}_-^{(t)} + \sum_{n'=1}^{N}\sum_{l=2}^{M}\alpha_{n,i,n',l}^{(t)}\boldsymbol{k}_{n',l}^{(t)}\right)$$

$$\cdot \left(\beta_{\overline{n},j,+}^{(t)}\boldsymbol{q}_+^{(t)\top} + \beta_{\overline{n},j,-}^{(t)}\boldsymbol{q}_-^{(t)\top} + \sum_{n'=1}^{N}\sum_{l=2}^{M}\beta_{\overline{n},j,n',l}^{(t)}\boldsymbol{q}_{n',l}^{(t)\top}\right)$$

for $n \neq \overline{n}$,

$$\langle \boldsymbol{q}_-^{(t+1)}, \boldsymbol{k}_+^{(t+1)}\rangle - \langle \boldsymbol{q}_-^{(t)}, \boldsymbol{k}_+^{(t)}\rangle$$

$$= \langle \Delta\boldsymbol{q}_-^{(t)}, \boldsymbol{k}_+^{(t)}\rangle + \langle \boldsymbol{q}_-^{(t)}, \Delta\boldsymbol{k}_+^{(t)}\rangle + \langle \Delta\boldsymbol{q}_-^{(t)}, \Delta\boldsymbol{k}_+^{(t)}\rangle$$

$$= \alpha_{-,-}^{(t)}\langle \boldsymbol{k}_+^{(t)}, \boldsymbol{k}_-^{(t)}\rangle + \sum_{n\in S_-}\sum_{i=2}^{M}\alpha_{n,-,i}^{(t)}\langle \boldsymbol{k}_{n,i}^{(t)}, \boldsymbol{k}_+^{(t)}\rangle$$

$$+ \beta_{+,+}^{(t)}\langle \boldsymbol{q}_+^{(t)}, \boldsymbol{q}_-^{(t)}\rangle + \sum_{n\in S_+}\sum_{i=2}^{M}\beta_{n,+,i}^{(t)}\langle \boldsymbol{q}_{n,i}^{(t)}, \boldsymbol{q}_-^{(t)}\rangle \qquad (197)$$

$$+ \left(\alpha_{-,-}^{(t)}\boldsymbol{k}_-^{(t)} + \sum_{n\in S_-}\sum_{i=2}^{M}\alpha_{n,-,i}^{(t)}\boldsymbol{k}_{n,i}^{(t)}\right)$$

$$\cdot \left(\beta_{+,+}^{(t)}\boldsymbol{q}_+^{(t)\top} + \sum_{n\in S_+}\sum_{i=2}^{M}\beta_{n,+,i}^{(t)}\boldsymbol{q}_{n,i}^{(t)\top}\right),$$

$$
\begin{aligned}
&\langle \boldsymbol{q}_+^{(t+1)}, \boldsymbol{q}_-^{(t+1)} \rangle - \langle \boldsymbol{q}_+^{(t)}, \boldsymbol{q}_-^{(t)} \rangle \\
&= \langle \Delta\boldsymbol{q}_+^{(t)}, \boldsymbol{q}_-^{(t)} \rangle + \langle \boldsymbol{q}_+^{(t)}, \Delta\boldsymbol{q}_-^{(t)} \rangle + \langle \Delta\boldsymbol{q}_+^{(t)}, \Delta\boldsymbol{q}_-^{(t)} \rangle \\
&= \alpha_{+,+}^{(t)} \langle \boldsymbol{q}_-^{(t)}, \boldsymbol{k}_+^{(t)} \rangle + \sum_{n \in S_+} \sum_{i=2}^{M} \alpha_{n,+,i}^{(t)} \langle \boldsymbol{q}_-^{(t)}, \boldsymbol{k}_{n,i}^{(t)} \rangle \\
&\quad + \alpha_{-,-}^{(t)} \langle \boldsymbol{q}_+^{(t)}, \boldsymbol{k}_-^{(t)} \rangle + \sum_{n \in S_-} \sum_{i=2}^{M} \alpha_{n,-,i}^{(t)} \langle \boldsymbol{q}_+^{(t)}, \boldsymbol{k}_{n,i}^{(t)} \rangle \\
&\quad + \Big( \alpha_{+,+}^{(t)} \boldsymbol{k}_+^{(t)} + \sum_{n \in S_+} \sum_{i=2}^{M} \alpha_{n,+,i}^{(t)} \boldsymbol{k}_{n,i}^{(t)} \Big) \\
&\quad \cdot \Big( \alpha_{-,-}^{(t)} \boldsymbol{k}_-^{(t)\top} + \sum_{n \in S_-} \sum_{i=2}^{M} \alpha_{n,-,i}^{(t)} \boldsymbol{k}_{n,i}^{(t)\top} \Big),
\end{aligned}
\tag{198}
$$

$$
\begin{aligned}
&\|\boldsymbol{q}_+^{(t+1)}\|_2^2 - \|\boldsymbol{q}_+^{(t)}\|_2^2 \\
&= 2\langle \Delta\boldsymbol{q}_+^{(t)}, \boldsymbol{q}_+^{(t)} \rangle + \langle \Delta\boldsymbol{q}_+^{(t)}, \Delta\boldsymbol{q}_+^{(t)} \rangle \\
&= 2\alpha_{+,+}^{(t)} \langle \boldsymbol{q}_+^{(t)}, \boldsymbol{k}_+^{(t)} \rangle + 2 \sum_{n \in S_+} \sum_{i=2}^{M} \alpha_{n,+,i}^{(t)} \langle \boldsymbol{q}_+^{(t)}, \boldsymbol{k}_{n,i}^{(t)} \rangle \\
&\quad + \Big( \alpha_{+,+}^{(t)} \boldsymbol{k}_+^{(t)} + \sum_{n \in S_+} \sum_{i=2}^{M} \alpha_{n,+,i}^{(t)} \boldsymbol{k}_{n,i}^{(t)} \Big) \\
&\quad \cdot \Big( \alpha_{+,+}^{(t)} \boldsymbol{k}_+^{(t)\top} + \sum_{n \in S_+} \sum_{i=2}^{M} \alpha_{n,+,i}^{(t)} \boldsymbol{k}_{n,i}^{(t)\top} \Big),
\end{aligned}
\tag{199}
$$

$$
\begin{aligned}
&\|\boldsymbol{q}_-^{(t+1)}\|_2^2 - \|\boldsymbol{q}_-^{(t)}\|_2^2 \\
&= 2\langle \Delta\boldsymbol{q}_-^{(t)}, \boldsymbol{q}_-^{(t)} \rangle + \langle \Delta\boldsymbol{q}_-^{(t)}, \Delta\boldsymbol{q}_-^{(t)} \rangle \\
&= 2\alpha_{-,-}^{(t)} \langle \boldsymbol{q}_-^{(t)}, \boldsymbol{k}_-^{(t)} \rangle + 2 \sum_{n \in S_-} \sum_{i=2}^{M} \alpha_{n,-,i}^{(t)} \langle \boldsymbol{q}_-^{(t)}, \boldsymbol{k}_{n,i}^{(t)} \rangle \\
&\quad + \Big( \alpha_{-,-}^{(t)} \boldsymbol{k}_-^{(t)} + \sum_{n \in S_-} \sum_{i=2}^{M} \alpha_{n,-,i}^{(t)} \boldsymbol{k}_{n,i}^{(t)} \Big) \\
&\quad \cdot \Big( \alpha_{-,-}^{(t)} \boldsymbol{k}_-^{(t)\top} + \sum_{n \in S_-} \sum_{i=2}^{M} \alpha_{n,-,i}^{(t)} \boldsymbol{k}_{n,i}^{(t)\top} \Big),
\end{aligned}
\tag{200}
$$

$$
\begin{aligned}
&\|\boldsymbol{q}_{n,i}^{(t+1)}\|_2^2 - \|\boldsymbol{q}_{n,i}^{(t)}\|_2^2 \\
&= 2\langle \Delta\boldsymbol{q}_{n,i}^{(t)}, \boldsymbol{q}_{n,i}^{(t)} \rangle + \langle \Delta\boldsymbol{q}_{n,i}^{(t)}, \Delta\boldsymbol{q}_{n,i}^{(t)} \rangle \\
&= 2\alpha_{n,i,+}^{(t)} \langle \boldsymbol{q}_{n,i}^{(t)}, \boldsymbol{k}_+^{(t)} \rangle + 2\alpha_{n,i,-}^{(t)} \langle \boldsymbol{q}_{n,i}^{(t)}, \boldsymbol{k}_-^{(t)} \rangle + 2 \sum_{n'=1}^{N} \sum_{l=2}^{M} \alpha_{n,i,n',l}^{(t)} \langle \boldsymbol{q}_{n,i}^{(t)}, \boldsymbol{k}_{n',l}^{(t)} \rangle \\
&\quad + \Big( \alpha_{n,i,+}^{(t)} \boldsymbol{k}_+^{(t)} + \alpha_{n,i,-}^{(t)} \boldsymbol{k}_-^{(t)} + \sum_{n'=1}^{N} \sum_{l=2}^{M} \alpha_{n,i,n',l}^{(t)} \boldsymbol{k}_{n',l}^{(t)} \Big) \\
&\quad \cdot \Big( \alpha_{n,i,+}^{(t)} \boldsymbol{k}_+^{(t)\top} + \alpha_{n,i,-}^{(t)} \boldsymbol{k}_-^{(t)\top} + \sum_{n'=1}^{N} \sum_{l=2}^{M} \alpha_{n,i,n',l}^{(t)} \boldsymbol{k}_{n',l}^{(t)\top} \Big),
\end{aligned}
\tag{201}
$$

$$\langle \boldsymbol{q}_+^{(t+1)}, \boldsymbol{q}_{n,i}^{(t+1)} \rangle - \langle \boldsymbol{q}_+^{(t)}, \boldsymbol{q}_{n,i}^{(t)} \rangle$$

$$= \langle \Delta \boldsymbol{q}_+^{(t)}, \boldsymbol{q}_{n,i}^{(t)} \rangle + \langle \boldsymbol{q}_+^{(t)}, \Delta \boldsymbol{q}_{n,i}^{(t)} \rangle + \langle \Delta \boldsymbol{q}_+^{(t)}, \Delta \boldsymbol{q}_{n,i}^{(t)} \rangle$$

$$= \alpha_{+,+}^{(t)} \langle \boldsymbol{q}_{n,i}^{(t)}, \boldsymbol{k}_+^{(t)} \rangle + \sum_{n' \in S_+} \sum_{l=2}^{M} \alpha_{n',+,l}^{(t)} \langle \boldsymbol{q}_{n,i}^{(t)}, \boldsymbol{k}_{n',l}^{(t)} \rangle$$

$$+ \alpha_{n,i,+}^{(t)} \langle \boldsymbol{q}_+^{(t)}, \boldsymbol{k}_+^{(t)} \rangle + \alpha_{n,i,-}^{(t)} \langle \boldsymbol{q}_+^{(t)}, \boldsymbol{k}_-^{(t)} \rangle + \sum_{n'=1}^{N} \sum_{l=2}^{M} \alpha_{n,i,n',l}^{(t)} \langle \boldsymbol{q}_+^{(t)}, \boldsymbol{k}_{n',l}^{(t)} \rangle \qquad (202)$$

$$+ \left( \alpha_{+,+}^{(t)} \boldsymbol{k}_+^{(t)} + \sum_{n \in S_+} \sum_{i=2}^{M} \alpha_{n,+,i}^{(t)} \boldsymbol{k}_{n,i}^{(t)} \right)$$

$$\cdot \left( \alpha_{n,i,+}^{(t)} \boldsymbol{k}_+^{(t)\top} + \alpha_{n,i,-}^{(t)} \boldsymbol{k}_-^{(t)\top} + \sum_{n'=1}^{N} \sum_{l=2}^{M} \alpha_{n,i,n',l}^{(t)} \boldsymbol{k}_{n',l}^{(t)\top} \right),$$

$$\langle \boldsymbol{q}_-^{(t+1)}, \boldsymbol{q}_{n,i}^{(t+1)} \rangle - \langle \boldsymbol{q}_-^{(t)}, \boldsymbol{q}_{n,i}^{(t)} \rangle$$

$$= \langle \Delta \boldsymbol{q}_-^{(t)}, \boldsymbol{q}_{n,i}^{(t)} \rangle + \langle \boldsymbol{q}_-^{(t)}, \Delta \boldsymbol{q}_{n,i}^{(t)} \rangle + \langle \Delta \boldsymbol{q}_-^{(t)}, \Delta \boldsymbol{q}_{n,i}^{(t)} \rangle$$

$$= \alpha_{-,-}^{(t)} \langle \boldsymbol{q}_{n,i}^{(t)}, \boldsymbol{k}_-^{(t)} \rangle + \sum_{n' \in S_-} \sum_{l=2}^{M} \alpha_{n',-,l}^{(t)} \langle \boldsymbol{q}_{n,i}^{(t)}, \boldsymbol{k}_{n',l}^{(t)} \rangle$$

$$+ \alpha_{n,i,+}^{(t)} \langle \boldsymbol{q}_-^{(t)}, \boldsymbol{k}_+^{(t)} \rangle + \alpha_{n,i,-}^{(t)} \langle \boldsymbol{q}_-^{(t)}, \boldsymbol{k}_-^{(t)} \rangle + \sum_{n'=1}^{N} \sum_{l=2}^{M} \alpha_{n,i,n',l}^{(t)} \langle \boldsymbol{q}_-^{(t)}, \boldsymbol{k}_{n',l}^{(t)} \rangle \qquad (203)$$

$$+ \left( \alpha_{-,-}^{(t)} \boldsymbol{k}_-^{(t)} + \sum_{n \in S_-} \sum_{i=2}^{M} \alpha_{n,-,i}^{(t)} \boldsymbol{k}_{n,i}^{(t)} \right)$$

$$\cdot \left( \alpha_{n,i,+}^{(t)} \boldsymbol{k}_+^{(t)\top} + \alpha_{n,i,-}^{(t)} \boldsymbol{k}_-^{(t)\top} + \sum_{n'=1}^{N} \sum_{l=2}^{M} \alpha_{n,i,n',l}^{(t)} \boldsymbol{k}_{n',l}^{(t)\top} \right),$$

$$\langle \boldsymbol{q}_{n,i}^{(t+1)}, \boldsymbol{q}_{n,j}^{(t+1)} \rangle - \langle \boldsymbol{q}_{n,i}^{(t)}, \boldsymbol{q}_{n,j}^{(t)} \rangle$$

$$= \langle \Delta \boldsymbol{q}_{n,i}^{(t)}, \boldsymbol{q}_{n,j}^{(t)} \rangle + \langle \boldsymbol{q}_{n,i}^{(t)}, \Delta \boldsymbol{q}_{n,j}^{(t)} \rangle + \langle \Delta \boldsymbol{q}_{n,i}^{(t)}, \Delta \boldsymbol{q}_{n,j}^{(t)} \rangle$$

$$= \alpha_{n,i,+}^{(t)} \langle \boldsymbol{q}_{n,j}^{(t)}, \boldsymbol{k}_+^{(t)} \rangle + \alpha_{n,i,-}^{(t)} \langle \boldsymbol{q}_{n,j}^{(t)}, \boldsymbol{k}_-^{(t)} \rangle + \sum_{n'=1}^{N} \sum_{l=2}^{M} \alpha_{n,i,n',l}^{(t)} \langle \boldsymbol{q}_{n,j}^{(t)}, \boldsymbol{k}_{n',l}^{(t)} \rangle$$

$$+ \alpha_{n,j,+}^{(t)} \langle \boldsymbol{q}_{n,i}^{(t)}, \boldsymbol{k}_+^{(t)} \rangle + \alpha_{n,j,-}^{(t)} \langle \boldsymbol{q}_{n,i}^{(t)}, \boldsymbol{k}_-^{(t)} \rangle + \sum_{n'=1}^{N} \sum_{l=2}^{M} \alpha_{n,j,n',l}^{(t)} \langle \boldsymbol{q}_{n,i}^{(t)}, \boldsymbol{k}_{n',l}^{(t)} \rangle \qquad (204)$$

$$+ \left( \alpha_{n,i,+}^{(t)} \boldsymbol{k}_+^{(t)} + \alpha_{n,i,-}^{(t)} \boldsymbol{k}_-^{(t)} + \sum_{n'=1}^{N} \sum_{l=2}^{M} \alpha_{n,i,n',l}^{(t)} \boldsymbol{k}_{n',l}^{(t)} \right)$$

$$\cdot \left( \alpha_{n,j,+}^{(t)} \boldsymbol{k}_+^{(t)\top} + \alpha_{n,j,-}^{(t)} \boldsymbol{k}_-^{(t)\top} + \sum_{n'=1}^{N} \sum_{l=2}^{M} \alpha_{n,j,n',l}^{(t)} \boldsymbol{k}_{n',l}^{(t)\top} \right),$$

$$\langle \boldsymbol{q}_{n,i}^{(t+1)}, \boldsymbol{q}_{\overline{n},j}^{(t+1)} \rangle - \langle \boldsymbol{q}_{n,i}^{(t)}, \boldsymbol{q}_{\overline{n},j}^{(t)} \rangle$$

$$= \langle \Delta \boldsymbol{q}_{n,i}^{(t)}, \boldsymbol{q}_{\overline{n},j}^{(t)} \rangle + \langle \boldsymbol{q}_{n,i}^{(t)}, \Delta \boldsymbol{q}_{\overline{n},j}^{(t)} \rangle + \langle \Delta \boldsymbol{q}_{n,i}^{(t)}, \Delta \boldsymbol{q}_{\overline{n},j}^{(t)} \rangle$$

$$= \alpha_{n,i,+}^{(t)} \langle \boldsymbol{q}_{\overline{n},j}^{(t)}, \boldsymbol{k}_{+}^{(t)} \rangle + \alpha_{n,i,-}^{(t)} \langle \boldsymbol{q}_{\overline{n},j}^{(t)}, \boldsymbol{k}_{-}^{(t)} \rangle + \sum_{n'=1}^{N} \sum_{l=2}^{M} \alpha_{n,i,n',l}^{(t)} \langle \boldsymbol{q}_{\overline{n},j}^{(t)}, \boldsymbol{k}_{n',l}^{(t)} \rangle$$

$$+ \alpha_{\overline{n},j,+}^{(t)} \langle \boldsymbol{q}_{n,i}^{(t)}, \boldsymbol{k}_{+}^{(t)} \rangle + \alpha_{\overline{n},j,-}^{(t)} \langle \boldsymbol{q}_{n,i}^{(t)}, \boldsymbol{k}_{-}^{(t)} \rangle + \sum_{n'=1}^{N} \sum_{l=2}^{M} \alpha_{\overline{n},j,n',l}^{(t)} \langle \boldsymbol{q}_{n,i}^{(t)}, \boldsymbol{k}_{n',l}^{(t)} \rangle \qquad (205)$$

$$+ \left( \alpha_{n,i,+}^{(t)} \boldsymbol{k}_{+}^{(t)} + \alpha_{n,i,-}^{(t)} \boldsymbol{k}_{-}^{(t)} + \sum_{n'=1}^{N} \sum_{l=2}^{M} \alpha_{n,i,n',l}^{(t)} \boldsymbol{k}_{n',l}^{(t)} \right)$$

$$\cdot \left( \alpha_{\overline{n},j,+}^{(t)} \boldsymbol{k}_{+}^{(t)\top} + \alpha_{\overline{n},j,-}^{(t)} \boldsymbol{k}_{-}^{(t)\top} + \sum_{n'=1}^{N} \sum_{l=2}^{M} \alpha_{\overline{n},j,n',l}^{(t)} \boldsymbol{k}_{n',l}^{(t)\top} \right)$$

for $n \neq \overline{n}$,

$$\|\boldsymbol{k}_{+}^{(t+1)}\|_2^2 - \|\boldsymbol{k}_{+}^{(t)}\|_2^2$$

$$= 2\langle \Delta \boldsymbol{k}_{+}^{(t)}, \boldsymbol{k}_{+}^{(t)} \rangle + \langle \Delta \boldsymbol{k}_{+}^{(t)}, \Delta \boldsymbol{k}_{+}^{(t)} \rangle$$

$$= 2\beta_{+,+}^{(t)} \langle \boldsymbol{q}_{+}^{(t)}, \boldsymbol{k}_{+}^{(t)} \rangle + 2 \sum_{n \in S_+} \sum_{i=2}^{M} \beta_{n,+,i}^{(t)} \langle \boldsymbol{q}_{n,i}^{(t)}, \boldsymbol{k}_{+}^{(t)} \rangle$$

$$+ \left( \beta_{+,+}^{(t)} \boldsymbol{q}_{+}^{(t)} + \sum_{n \in S_+} \sum_{i=2}^{M} \beta_{n,+,i}^{(t)} \boldsymbol{q}_{n,i}^{(t)} \right) \qquad (206)$$

$$\cdot \left( \beta_{+,+}^{(t)} \boldsymbol{q}_{+}^{(t)\top} + \sum_{n \in S_+} \sum_{i=2}^{M} \beta_{n,+,i}^{(t)} \boldsymbol{q}_{n,i}^{(t)\top} \right),$$

$$\|\boldsymbol{k}_{-}^{(t+1)}\|_2^2 - \|\boldsymbol{k}_{-}^{(t)}\|_2^2$$

$$= 2\langle \Delta \boldsymbol{k}_{-}^{(t)}, \boldsymbol{k}_{-}^{(t)} \rangle + \langle \Delta \boldsymbol{k}_{-}^{(t)}, \Delta \boldsymbol{k}_{-}^{(t)} \rangle$$

$$= 2\beta_{-,-}^{(t)} \langle \boldsymbol{q}_{-}^{(t)}, \boldsymbol{k}_{-}^{(t)} \rangle + 2 \sum_{n \in S_-} \sum_{i=2}^{M} \beta_{n,-,i}^{(t)} \langle \boldsymbol{q}_{n,i}^{(t)}, \boldsymbol{k}_{-}^{(t)} \rangle$$

$$+ \left( \beta_{-,-}^{(t)} \boldsymbol{q}_{-}^{(t)} + \sum_{n \in S_-} \sum_{i=2}^{M} \beta_{n,-,i}^{(t)} \boldsymbol{q}_{n,i}^{(t)} \right) \qquad (207)$$

$$\cdot \left( \beta_{-,-}^{(t)} \boldsymbol{q}_{-}^{(t)\top} + \sum_{n \in S_-} \sum_{i=2}^{M} \beta_{n,-,i}^{(t)} \boldsymbol{q}_{n,i}^{(t)\top} \right),$$

$$\|\boldsymbol{k}_{n,i}^{(t+1)}\|_2^2 - \|\boldsymbol{k}_{n,i}^{(t)}\|_2^2$$
$$= 2\langle \Delta \boldsymbol{k}_{n,i}^{(t)}, \boldsymbol{k}_{n,i}^{(t)}\rangle + \langle \Delta \boldsymbol{k}_{n,i}^{(t)}, \Delta \boldsymbol{k}_{n,i}^{(t)}\rangle$$
$$= 2\beta_{n,i,+}^{(t)}\langle \boldsymbol{q}_+^{(t)}, \boldsymbol{k}_{n,i}^{(t)}\rangle + 2\beta_{n,i,-}^{(t)}\langle \boldsymbol{q}_-^{(t)}, \boldsymbol{k}_{n,i}^{(t)}\rangle + 2\sum_{n'=1}^{N}\sum_{l=2}^{M}\beta_{n,i,n',l}^{(t)}\langle \boldsymbol{q}_{n',l}^{(t)}, \boldsymbol{k}_{n,i}^{(t)}\rangle$$
$$+ \left(\beta_{n,i,+}^{(t)}\boldsymbol{q}_+^{(t)} + \beta_{n,i,-}^{(t)}\boldsymbol{q}_-^{(t)} + \sum_{n'=1}^{N}\sum_{l=2}^{M}\beta_{n,i,n',l}^{(t)}\boldsymbol{q}_{n',l}^{(t)}\right)$$
$$\cdot \left(\beta_{n,i,+}^{(t)}\boldsymbol{q}_+^{(t)\top} + \beta_{n,i,-}^{(t)}\boldsymbol{q}_-^{(t)\top} + \sum_{n'=1}^{N}\sum_{l=2}^{M}\beta_{n,i,n',l}^{(t)}\boldsymbol{q}_{n',l}^{(t)\top}\right),$$

(208)

$$\langle \boldsymbol{k}_+^{(t+1)}, \boldsymbol{k}_{n,i}^{(t+1)}\rangle - \langle \boldsymbol{k}_+^{(t)}, \boldsymbol{k}_{n,i}^{(t)}\rangle$$
$$= \langle \Delta \boldsymbol{k}_+^{(t)}, \boldsymbol{k}_{n,i}^{(t)}\rangle + \langle \boldsymbol{k}_+^{(t)}, \Delta \boldsymbol{k}_{n,i}^{(t)}\rangle + \langle \Delta \boldsymbol{k}_+^{(t)}, \Delta \boldsymbol{k}_{n,i}^{(t)}\rangle$$
$$= \beta_{+,+}^{(t)}\langle \boldsymbol{q}_+^{(t)}, \boldsymbol{k}_{n,i}^{(t)}\rangle + \sum_{n'\in S_+}\sum_{l=2}^{M}\beta_{n',+,l}^{(t)}\langle \boldsymbol{q}_{n',l}^{(t)}, \boldsymbol{k}_{n,i}^{(t)}\rangle$$
$$+ \beta_{n,i,+}^{(t)}\langle \boldsymbol{q}_+^{(t)}, \boldsymbol{k}_+^{(t)}\rangle + \beta_{n,i,-}^{(t)}\langle \boldsymbol{q}_-^{(t)}, \boldsymbol{k}_+^{(t)}\rangle + \sum_{n'=1}^{N}\sum_{l=2}^{M}\beta_{n,i,n',l}^{(t)}\langle \boldsymbol{q}_{n',l}^{(t)}, \boldsymbol{k}_+^{(t)}\rangle$$
$$+ \left(\beta_{+,+}^{(t)}\boldsymbol{q}_+^{(t)} + \sum_{n'\in S_+}\sum_{l=2}^{M}\beta_{n',+,l}^{(t)}\boldsymbol{q}_{n',l}^{(t)}\right)$$
$$\cdot \left(\beta_{n,i,+}^{(t)}\boldsymbol{q}_+^{(t)\top} + \beta_{n,i,-}^{(t)}\boldsymbol{q}_-^{(t)\top} + \sum_{n'=1}^{N}\sum_{l=2}^{M}\beta_{n,i,n',l}^{(t)}\boldsymbol{q}_{n',l}^{(t)\top}\right),$$

(209)

$$\langle \boldsymbol{k}_-^{(t+1)}, \boldsymbol{k}_{n,i}^{(t+1)}\rangle - \langle \boldsymbol{k}_-^{(t)}, \boldsymbol{k}_{n,i}^{(t)}\rangle$$
$$= \langle \Delta \boldsymbol{k}_-^{(t)}, \boldsymbol{k}_{n,i}^{(t)}\rangle + \langle \boldsymbol{k}_-^{(t)}, \Delta \boldsymbol{k}_{n,i}^{(t)}\rangle + \langle \Delta \boldsymbol{k}_-^{(t)}, \Delta \boldsymbol{k}_{n,i}^{(t)}\rangle$$
$$= \beta_{-,-}^{(t)}\langle \boldsymbol{q}_-^{(t)}, \boldsymbol{k}_{n,i}^{(t)}\rangle + \sum_{n'\in S_-}\sum_{l=2}^{M}\beta_{n',-,l}^{(t)}\langle \boldsymbol{q}_{n',l}^{(t)}, \boldsymbol{k}_{n,i}^{(t)}\rangle$$
$$+ \beta_{n,i,+}^{(t)}\langle \boldsymbol{q}_+^{(t)}, \boldsymbol{k}_-^{(t)}\rangle + \beta_{n,i,-}^{(t)}\langle \boldsymbol{q}_-^{(t)}, \boldsymbol{k}_-^{(t)}\rangle + \sum_{n'=1}^{N}\sum_{l=2}^{M}\beta_{n,i,n',l}^{(t)}\langle \boldsymbol{q}_{n',l}^{(t)}, \boldsymbol{k}_+^{(t)}\rangle$$
$$+ \left(\beta_{-,-}^{(t)}\boldsymbol{q}_-^{(t)} + \sum_{n'\in S_-}\sum_{l=2}^{M}\beta_{n',-,l}^{(t)}\boldsymbol{q}_{n',l}^{(t)}\right)$$
$$\cdot \left(\beta_{n,i,+}^{(t)}\boldsymbol{q}_+^{(t)\top} + \beta_{n,i,-}^{(t)}\boldsymbol{q}_-^{(t)\top} + \sum_{n'=1}^{N}\sum_{l=2}^{M}\beta_{n,i,n',l}^{(t)}\boldsymbol{q}_{n',l}^{(t)\top}\right),$$

(210)

$$\langle \boldsymbol{k}_{n,i}^{(t+1)}, \boldsymbol{k}_{n,j}^{(t+1)}\rangle - \langle \boldsymbol{k}_{n,i}^{(t)}, \boldsymbol{k}_{n,j}^{(t)}\rangle$$

$$= \langle \Delta\boldsymbol{k}_{n,i}^{(t)}, \boldsymbol{k}_{n,j}^{(t)}\rangle + \langle \boldsymbol{k}_{n,i}^{(t)}, \Delta\boldsymbol{k}_{n,j}^{(t)}\rangle + \langle \Delta\boldsymbol{k}_{n,i}^{(t)}, \Delta\boldsymbol{k}_{n,j}^{(t)}\rangle$$

$$= \beta_{n,i,+}^{(t)}\langle \boldsymbol{q}_+^{(t)}, \boldsymbol{k}_{n,j}^{(t)}\rangle + \beta_{n,i,-}^{(t)}\langle \boldsymbol{q}_-^{(t)}, \boldsymbol{k}_{n,j}^{(t)}\rangle + \sum_{n'=1}^{N}\sum_{l=2}^{M}\beta_{n,i,n',l}^{(t)}\langle \boldsymbol{q}_{n',l}^{(t)}, \boldsymbol{k}_{n,j}^{(t)}\rangle$$

$$+ \beta_{n,j,+}^{(t)}\langle \boldsymbol{q}_+^{(t)}, \boldsymbol{k}_{n,i}^{(t)}\rangle + \beta_{n,j,-}^{(t)}\langle \boldsymbol{q}_-^{(t)}, \boldsymbol{k}_{n,i}^{(t)}\rangle + \sum_{n'=1}^{N}\sum_{l=2}^{M}\beta_{n,j,n',l}^{(t)}\langle \boldsymbol{q}_{n',l}^{(t)}, \boldsymbol{k}_{n,i}^{(t)}\rangle \qquad (211)$$

$$+ \left(\beta_{n,i,+}^{(t)}\boldsymbol{q}_+^{(t)} + \beta_{n,i,-}^{(t)}\boldsymbol{q}_-^{(t)} + \sum_{n'=1}^{N}\sum_{l=2}^{M}\beta_{n,i,n',l}^{(t)}\boldsymbol{q}_{n',l}^{(t)}\right)$$

$$\cdot \left(\beta_{n,j,+}^{(t)}\boldsymbol{q}_+^{(t)\top} + \beta_{n,j,-}^{(t)}\boldsymbol{q}_-^{(t)\top} + \sum_{n'=1}^{N}\sum_{l=2}^{M}\beta_{n,j,n',l}^{(t)}\boldsymbol{q}_{n',l}^{(t)\top}\right),$$

$$\langle \boldsymbol{k}_{n,i}^{(t+1)}, \boldsymbol{k}_{\overline{n},j}^{(t+1)}\rangle - \langle \boldsymbol{k}_{n,i}^{(t)}, \boldsymbol{k}_{\overline{n},j}^{(t)}\rangle$$

$$= \langle \Delta\boldsymbol{k}_{n,i}^{(t)}, \boldsymbol{k}_{\overline{n},j}^{(t)}\rangle + \langle \boldsymbol{k}_{n,i}^{(t)}, \Delta\boldsymbol{k}_{\overline{n},j}^{(t)}\rangle + \langle \Delta\boldsymbol{k}_{n,i}^{(t)}, \Delta\boldsymbol{k}_{\overline{n},j}^{(t)}\rangle$$

$$= \beta_{n,i,+}^{(t)}\langle \boldsymbol{q}_+^{(t)}, \boldsymbol{k}_{\overline{n},j}^{(t)}\rangle + \beta_{n,i,-}^{(t)}\langle \boldsymbol{q}_-^{(t)}, \boldsymbol{k}_{\overline{n},j}^{(t)}\rangle + \sum_{n'=1}^{N}\sum_{l=2}^{M}\beta_{n,i,n',l}^{(t)}\langle \boldsymbol{q}_{n',l}^{(t)}, \boldsymbol{k}_{\overline{n},j}^{(t)}\rangle$$

$$+ \beta_{\overline{n},j,+}^{(t)}\langle \boldsymbol{q}_+^{(t)}, \boldsymbol{k}_{n,i}^{(t)}\rangle + \beta_{\overline{n},j,-}^{(t)}\langle \boldsymbol{q}_-^{(t)}, \boldsymbol{k}_{n,i}^{(t)}\rangle + \sum_{n'=1}^{N}\sum_{l=2}^{M}\beta_{\overline{n},j,n',l}^{(t)}\langle \boldsymbol{q}_{n',l}^{(t)}, \boldsymbol{k}_{n,i}^{(t)}\rangle \qquad (212)$$

$$+ \left(\beta_{n,i,+}^{(t)}\boldsymbol{q}_+^{(t)} + \beta_{n,i,-}^{(t)}\boldsymbol{q}_-^{(t)} + \sum_{n'=1}^{N}\sum_{l=2}^{M}\beta_{n,i,n',l}^{(t)}\boldsymbol{q}_{n',l}^{(t)}\right)$$

$$\cdot \left(\beta_{\overline{n},j,+}^{(t)}\boldsymbol{q}_+^{(t)\top} + \beta_{\overline{n},j,-}^{(t)}\boldsymbol{q}_-^{(t)\top} + \sum_{n'=1}^{N}\sum_{l=2}^{M}\beta_{\overline{n},j,n',l}^{(t)}\boldsymbol{q}_{n',l}^{(t)\top}\right)$$

for $n \neq \overline{n}$,

$$\langle \boldsymbol{q}_+^{(t+1)}, \boldsymbol{k}_-^{(t+1)}\rangle - \langle \boldsymbol{q}_+^{(t)}, \boldsymbol{k}_-^{(t)}\rangle$$

$$= \langle \Delta\boldsymbol{q}_+^{(t)}, \boldsymbol{k}_-^{(t)}\rangle + \langle \boldsymbol{q}_+^{(t)}, \Delta\boldsymbol{k}_-^{(t)}\rangle + \langle \Delta\boldsymbol{q}_+^{(t)}, \Delta\boldsymbol{k}_-^{(t)}\rangle$$

$$= \alpha_{+,+}^{(t)}\langle \boldsymbol{k}_+^{(t)}, \boldsymbol{k}_-^{(t)}\rangle + \sum_{n\in S_+}\sum_{i=2}^{M}\alpha_{n,+,i}^{(t)}\langle \boldsymbol{k}_{n,i}^{(t)}, \boldsymbol{k}_-^{(t)}\rangle$$

$$+ \beta_{-,-}^{(t)}\langle \boldsymbol{q}_+^{(t)}, \boldsymbol{q}_-^{(t)}\rangle + \sum_{n\in S_-}\sum_{i=2}^{M}\beta_{n,-,i}^{(t)}\langle \boldsymbol{q}_{n,i}^{(t)}, \boldsymbol{q}_+^{(t)}\rangle \qquad (213)$$

$$+ \left(\alpha_{+,+}^{(t)}\boldsymbol{k}_+^{(t)} + \sum_{n\in S_+}\sum_{i=2}^{M}\alpha_{n,+,i}^{(t)}\boldsymbol{k}_{n,i}^{(t)}\right)$$

$$\cdot \left(\beta_{-,-}^{(t)}\boldsymbol{q}_-^{(t)\top} + \sum_{n\in S_-}\sum_{i=2}^{M}\beta_{n,-,i}^{(t)}\boldsymbol{q}_{n,i}^{(t)\top}\right),$$

$$\langle k_+^{(t+1)}, k_-^{(t+1)} \rangle - \langle k_+^{(t)}, k_-^{(t)} \rangle$$

$$= \langle \Delta k_+^{(t)}, k_-^{(t)} \rangle + \langle k_+^{(t)}, \Delta k_-^{(t)} \rangle + \langle \Delta k_+^{(t)}, \Delta k_-^{(t)} \rangle$$

$$= \beta_{+,+}^{(t)} \langle q_+^{(t)}, k_-^{(t)} \rangle + \sum_{n \in S_+} \sum_{i=2}^{M} \beta_{n,+,i}^{(t)} \langle q_{n,i}^{(t)}, k_-^{(t)} \rangle$$

$$+ \beta_{-,-}^{(t)} \langle q_-^{(t)}, k_+^{(t)} \rangle + \sum_{n \in S_-} \sum_{i=2}^{M} \beta_{n,-,i}^{(t)} \langle q_{n,i}^{(t)}, k_+^{(t)} \rangle \qquad (214)$$

$$+ \left( \beta_{+,+}^{(t)} q_+^{(t)} + \sum_{n \in S_+} \sum_{i=2}^{M} \beta_{n,+,i}^{(t)} q_{n,i}^{(t)} \right)$$

$$\cdot \left( \beta_{-,-}^{(t)} q_-^{(t)\top} + \sum_{n \in S_-} \sum_{i=2}^{M} \beta_{n,-,i}^{(t)} q_{n,i}^{(t)\top} \right).$$

### F.3  Proof of Lemma D.4

Let $T_0 = O(\frac{1}{\eta d_h^{\frac{1}{4}} \|\boldsymbol{\mu}\|_2^2 \|\boldsymbol{w}_O\|_2^2})$. By Lemma D.3, we have $|V_+^{(t)}|, |V_-^{(t)}|, |V_{n,i}^{(t)}| = O(d_h^{-\frac{1}{4}})$ for $t \in [0, T_0]$ by Lemma D.3. Plugging this into the expression for $\alpha$ and $\beta$ gives

$$|\alpha_{+,+}^{(t)}| = \left| \frac{\eta}{NM} \sum_{n \in S_+} -\ell_n'^{(t)} \|\boldsymbol{\mu}\|_2^2 \right.$$

$$\cdot \left( V_+^{(t)} \left( \frac{\exp(\langle q_+^{(t)}, k_+^{(t)} \rangle)}{\exp(\langle q_+^{(t)}, k_+^{(t)} \rangle) + \sum_{j=2}^{M} \exp(\langle q_+^{(t)}, k_{n,j}^{(t)} \rangle)} \right. \right.$$

$$- \left( \frac{\exp(\langle q_+^{(t)}, k_+^{(t)} \rangle)}{\exp(\langle q_+^{(t)}, k_+^{(t)} \rangle) + \sum_{j=2}^{M} \exp(\langle q_+^{(t)}, k_{n,j}^{(t)} \rangle)} \right)^2 \right)$$

$$- \sum_{i=2}^{M} \left( V_{n,i}^{(t)} \cdot \frac{\exp(\langle q_+^{(t)}, k_+^{(t)} \rangle)}{\exp(\langle q_+^{(t)}, k_+^{(t)} \rangle) + \sum_{j=2}^{M} \exp(\langle q_+^{(t)}, k_{n,j}^{(t)} \rangle)} \right.$$

$$\left. \left. \left. \cdot \frac{\exp(\langle q_+^{(t)}, k_{n,i}^{(t)} \rangle)}{\exp(\langle q_+^{(t)}, k_+^{(t)} \rangle) + \sum_{j=2}^{M} \exp(\langle q_+^{(t)}, k_{n,j}^{(t)} \rangle)} \right) \right) \right|$$

$$\le \frac{\eta \|\boldsymbol{\mu}\|_2^2}{NM} \cdot \frac{3NM}{4} \cdot O(d_h^{-\frac{1}{4}})$$

$$= O(\frac{\eta \|\boldsymbol{\mu}\|_2^2}{d_h^{\frac{1}{4}}}),$$

where the inequality is by $-\ell_n'^{(t)} \le 1$ and the property that attention is smaller than 1 ( e.g. $\frac{\exp(\langle q_+^{(t)}, k_+^{(t)} \rangle)}{\exp(\langle q_+^{(t)}, k_+^{(t)} \rangle) + \sum_{j=2}^{M} \exp(\langle q_+^{(t)}, k_{n,j}^{(t)} \rangle)} \le 1$ ). We also have

$$|\alpha_{n,+,i}^{(t)}| = \left| -\frac{\eta}{NM} \ell_n'^{(t)} \|\boldsymbol{\mu}\|_2^2 \right.$$

$$\cdot \left( - V_+^{(t)} \cdot \frac{\exp(\langle \boldsymbol{q}_+^{(t)}, \boldsymbol{k}_+^{(t)} \rangle)}{\exp(\langle \boldsymbol{q}_+^{(t)}, \boldsymbol{k}_+^{(t)} \rangle) + \sum\limits_{j=2}^{M} \exp(\langle \boldsymbol{q}_+^{(t)}, \boldsymbol{k}_{n,j}^{(t)} \rangle)} \right.$$

$$\cdot \frac{\exp(\langle \boldsymbol{q}_+^{(t)}, \boldsymbol{k}_{n,i}^{(t)} \rangle)}{\exp(\langle \boldsymbol{q}_+^{(t)}, \boldsymbol{k}_+^{(t)} \rangle) + \sum\limits_{j=2}^{M} \exp(\langle \boldsymbol{q}_+^{(t)}, \boldsymbol{k}_{n,j}^{(t)} \rangle)}$$

$$+ V_{n,i}^{(t)} \left( \frac{\exp(\langle \boldsymbol{q}_+^{(t)}, \boldsymbol{k}_{n,i}^{(t)} \rangle)}{\exp(\langle \boldsymbol{q}_+^{(t)}, \boldsymbol{k}_+^{(t)} \rangle) + \sum\limits_{j=2}^{M} \exp(\langle \boldsymbol{q}_+^{(t)}, \boldsymbol{k}_{n,j}^{(t)} \rangle)} \right.$$

$$\left. - \left( \frac{\exp(\langle \boldsymbol{q}_+^{(t)}, \boldsymbol{k}_{n,i}^{(t)} \rangle)}{\exp(\langle \boldsymbol{q}_+^{(t)}, \boldsymbol{k}_+^{(t)} \rangle) + \sum\limits_{j=2}^{M} \exp(\langle \boldsymbol{q}_+^{(t)}, \boldsymbol{k}_{n,j}^{(t)} \rangle)} \right)^2 \right)$$

$$- \sum_{k \neq i} \left( V_{n,k}^{(t)} \cdot \frac{\exp(\langle \boldsymbol{q}_+^{(t)}, \boldsymbol{k}_{n,i}^{(t)} \rangle)}{\exp(\langle \boldsymbol{q}_+^{(t)}, \boldsymbol{k}_+^{(t)} \rangle) + \sum\limits_{j=2}^{M} \exp(\langle \boldsymbol{q}_+^{(t)}, \boldsymbol{k}_{n,j}^{(t)} \rangle)} \right.$$

$$\left. \left. \cdot \frac{\exp(\langle \boldsymbol{q}_+^{(t)}, \boldsymbol{k}_{n,k}^{(t)} \rangle)}{\exp(\langle \boldsymbol{q}_+^{(t)}, \boldsymbol{k}_+^{(t)} \rangle) + \sum\limits_{j=2}^{M} \exp(\langle \boldsymbol{q}_+^{(t)}, \boldsymbol{k}_{n,j}^{(t)} \rangle)} \right) \right) \Bigg|$$

$$\leq \frac{\eta \|\boldsymbol{\mu}\|_2^2}{NM} \cdot M \cdot O(d_h^{-\frac{1}{4}})$$

$$= O\left( \frac{\eta \|\boldsymbol{\mu}\|_2^2}{d_h^{\frac{1}{4}} N} \right),$$

where the inequality is by $-\ell_n'^{(t)} \leq 1$ and the property that attention is smaller than 1. We also have

$$|\alpha_{n',i',+}^{(t)}| = \left| \frac{\eta}{NM} \sum_{n \in S_+} -\ell_n'^{(t)} \sum_{i=2}^{M} \langle \boldsymbol{\xi}_{n',i'}, \boldsymbol{\xi}_{n,i} \rangle \right.$$

$$\cdot \left( V_+^{(t)} \left( \frac{\exp(\langle \boldsymbol{q}_{n,i}^{(t)}, \boldsymbol{k}_+^{(t)} \rangle)}{\exp(\langle \boldsymbol{q}_{n,i}^{(t)}, \boldsymbol{k}_+^{(t)} \rangle) + \sum\limits_{j=2}^{M} \exp(\langle \boldsymbol{q}_{n,i}^{(t)}, \boldsymbol{k}_{n,j}^{(t)} \rangle)} \right. \right.$$

$$\left. - \left( \frac{\exp(\langle \boldsymbol{q}_{n,i}^{(t)}, \boldsymbol{k}_+^{(t)} \rangle)}{\exp(\langle \boldsymbol{q}_{n,i}^{(t)}, \boldsymbol{k}_+^{(t)} \rangle) + \sum\limits_{j=2}^{M} \exp(\langle \boldsymbol{q}_{n,i}^{(t)}, \boldsymbol{k}_{n,j}^{(t)} \rangle)} \right)^2 \right)$$

$$- \sum_{k=2}^{M} \left( V_{n,i}^{(t)} \cdot \frac{\exp(\langle \boldsymbol{q}_{n,i}^{(t)}, \boldsymbol{k}_+^{(t)} \rangle)}{\exp(\langle \boldsymbol{q}_{n,i}^{(t)}, \boldsymbol{k}_+^{(t)} \rangle) + \sum\limits_{j=2}^{M} \exp(\langle \boldsymbol{q}_{n,i}^{(t)}, \boldsymbol{k}_{n,j}^{(t)} \rangle)} \right.$$

$$\left. \left. \cdot \frac{\exp(\langle \boldsymbol{q}_{n,i}^{(t)}, \boldsymbol{k}_{n,k}^{(t)} \rangle)}{\exp(\langle \boldsymbol{q}_{n,i}^{(t)}, \boldsymbol{k}_+^{(t)} \rangle) + \sum\limits_{j=2}^{M} \exp(\langle \boldsymbol{q}_{n,i}^{(t)}, \boldsymbol{k}_{n,j}^{(t)} \rangle)} \right) \right) \right|$$

$$\leq \frac{\eta}{NM} \left( \frac{3\tilde{\sigma}_p^2 d}{2} + \frac{3N}{4} \cdot \tilde{\sigma}_p^2 \cdot \sqrt{d \log(4N^2 M^2/\delta)} \right) \cdot M \cdot O(d_h^{-\frac{1}{4}})$$

$$= O(\frac{\eta \sigma_p^2 d}{d_h^{\frac{1}{4}} N}),$$

where the inequality is by $-\ell_n'^{(t)} \leq 1$, Lemma C.4 and the property that attention is smaller than 1. Similarly, we have

$$|\alpha_{-,-}^{(t)}|, |\beta_{+,+}^{(t)}|, |\beta_{-,-}^{(t)}| = O\Big(\frac{\eta \|\boldsymbol{\mu}\|_2^2}{d_h^{\frac{1}{4}}}\Big),$$

$$|\alpha_{n,-,l}^{(t)}|, |\beta_{n,+,l}^{(t)}|, |\beta_{n,-,l}^{(t)}| = O\Big(\frac{\eta \|\boldsymbol{\mu}\|_2^2}{d_h^{\frac{1}{4}} N}\Big),$$

$$|\alpha_{n,l,-}^{(t)}|, |\beta_{n,l,+}^{(t)}|, |\beta_{n,l,-}^{(t)}|, |\alpha_{n,l,n',l'}^{(t)}|, |\beta_{n,l,n',l'}^{(t)}| = O\Big(\frac{\eta \sigma_p^2 d}{d_h^{\frac{1}{4}} N}\Big)$$

for $t \in [0, T_0]$.

Next we use induction to show that the following proposition $\mathcal{A}(t)$ holds for $t \in [0, T_0]$
$\mathcal{A}(t)$ :

$$|\langle \boldsymbol{q}_\pm^{(t)}, \boldsymbol{k}_\pm^{(t)}\rangle|, |\langle \boldsymbol{q}_{n,i}^{(t)}, \boldsymbol{k}_\pm^{(t)}\rangle|, |\langle \boldsymbol{q}_\pm^{(t)}, \boldsymbol{k}_{n,j}^{(t)}\rangle|, |\langle \boldsymbol{q}_{n,i}^{(t)}, \boldsymbol{k}_{n',j}^{(t)}\rangle|$$
$$= O\Big( \max\{\|\boldsymbol{\mu}\|_2^2, \sigma_p^2 d\} \cdot \sigma_h^2 \cdot \sqrt{d_h \log(6N^2 M^2/\delta)}\Big),$$

$$|\langle \boldsymbol{q}_\pm^{(t)}, \boldsymbol{q}_\mp^{(t)}\rangle|, |\langle \boldsymbol{q}_{n,i}^{(t)}, \boldsymbol{q}_\pm^{(t)}\rangle|, |\langle \boldsymbol{q}_{n,i}^{(t)}, \boldsymbol{q}_{n',j}^{(t)}\rangle|$$
$$= O\Big( \max\{\|\boldsymbol{\mu}\|_2^2, \sigma_p^2 d\} \cdot \sigma_h^2 \cdot \sqrt{d_h \log(6N^2 M^2/\delta)}\Big),$$

$$|\langle \boldsymbol{k}_\pm^{(t)}, \boldsymbol{k}_\mp^{(t)}\rangle|, |\langle \boldsymbol{k}_{n,i}^{(t)}, \boldsymbol{k}_\pm^{(t)}\rangle|, |\langle \boldsymbol{k}_{n,i}^{(t)}, \boldsymbol{k}_{n',j}^{(t)}\rangle|$$
$$= O\Big( \max\{\|\boldsymbol{\mu}\|_2^2, \sigma_p^2 d\} \cdot \sigma_h^2 \cdot \sqrt{d_h \log(6N^2 M^2/\delta)}\Big),$$

$$\|\boldsymbol{q}_\pm^{(t)}\|_2^2, \|\boldsymbol{k}_\pm^{(t)}\|_2^2 = \Theta(\|\boldsymbol{\mu}\|_2^2 \sigma_h^2 d_h)$$

$$\|\boldsymbol{q}_{n,i}^{(t)}\|_2^2, \|\boldsymbol{k}_{n,i}^{(t)}\|_2^2 = \Theta(\sigma_p^2 \sigma_h^2 d d_h)$$

for $i, j \in [M]\backslash\{1\}, n, n' \in [N]$.

By Lemma C.3 we know that $\mathcal{A}(0)$ is true. Now we assume $\mathcal{A}(0), \ldots, \mathcal{A}(T)$ is true, then we need to proof that $\mathcal{A}(T + 1)$ is true. We first proof $|\langle \boldsymbol{q}_+^{(T+1)}, \boldsymbol{k}_+^{(T+1)}\rangle| = O\Big( \max\{\|\boldsymbol{\mu}\|_2^2, \sigma_p^2 d\} \cdot \sigma_h^2 \cdot \sqrt{d_h \log(6N^2 M^2/\delta)}\Big)$, as an example.

$$\left| \langle \boldsymbol{q}_+^{(t+1)}, \boldsymbol{k}_+^{(t+1)} \rangle - \langle \boldsymbol{q}_+^{(t)}, \boldsymbol{k}_+^{(t)} \rangle \right|$$

$$= \left| \alpha_{+,+}^{(t)} \|\boldsymbol{k}_+^{(t)}\|_2^2 + \sum_{n \in S_+} \sum_{i=2}^{M} \alpha_{n,+,i}^{(t)} \langle \boldsymbol{k}_+^{(t)}, \boldsymbol{k}_{n,i}^{(t)} \rangle \right.$$

$$+ \beta_{+,+}^{(t)} \|\boldsymbol{q}_+^{(t)}\|_2^2 + \sum_{n \in S_+} \sum_{i=2}^{M} \beta_{n,+,i}^{(t)} \langle \boldsymbol{q}_+^{(t)}, \boldsymbol{q}_{n,i}^{(t)} \rangle$$

$$+ \left( \alpha_{+,+}^{(t)} \boldsymbol{k}_+^{(t)} + \sum_{n \in S_+} \sum_{i=2}^{M} \alpha_{n,+,i}^{(t)} \boldsymbol{k}_{n,i}^{(t)} \right)$$

$$\left. \cdot \left( \beta_{+,+}^{(t)} \boldsymbol{q}_+^{(t)\top} + \sum_{n \in S_+} \sum_{i=2}^{M} \beta_{n,+,i}^{(t)} \boldsymbol{q}_{n,i}^{(t)\top} \right) \right| \tag{215}$$

$$\leq O\left( \frac{\eta \|\boldsymbol{\mu}\|_2^2}{d_h^{\frac{1}{4}}} \right) \cdot \Theta(\|\boldsymbol{\mu}\|_2^2 \sigma_h^2 d_h)$$

$$+ NM \cdot O\left( \frac{\eta \|\boldsymbol{\mu}\|_2^2}{d_h^{\frac{1}{4}} N} \right) \cdot O\left( \max\{\|\boldsymbol{\mu}\|_2^2, \sigma_p^2 d\} \cdot \sigma_h^2 \cdot \sqrt{d_h \log(6N^2 M^2/\delta)} \right)$$

$$+ \{lower \ order \ term\}$$

$$= O\left( \eta \|\boldsymbol{\mu}\|_2^4 \sigma_h^2 d_h^{\frac{3}{4}} \right)$$

Taking a summation, we obtain that

$$|\langle \boldsymbol{q}_+^{(T+1)}, \boldsymbol{k}_+^{(T+1)} \rangle| \leq |\langle \boldsymbol{q}_+^{(0)}, \boldsymbol{k}_+^{(0)} \rangle| + \sum_{t=0}^{T} \left| \langle \boldsymbol{q}_+^{(t+1)}, \boldsymbol{k}_+^{(t+1)} \rangle - \langle \boldsymbol{q}_+^{(t)}, \boldsymbol{k}_+^{(t)} \rangle \right|$$

$$\leq |\langle \boldsymbol{q}_+^{(0)}, \boldsymbol{k}_+^{(0)} \rangle| + \sum_{t=0}^{T_0-1} \left| \langle \boldsymbol{q}_+^{(t+1)}, \boldsymbol{k}_+^{(t+1)} \rangle - \langle \boldsymbol{q}_+^{(t)}, \boldsymbol{k}_+^{(t)} \rangle \right|$$

$$\leq O\left( \max\{\|\boldsymbol{\mu}\|_2^2, \sigma_p^2 d\} \cdot \sigma_h^2 \cdot \sqrt{d_h \log(6N^2 M^2/\delta)} \right) + O\left( \frac{1}{\eta d_h^{\frac{1}{4}} \|\boldsymbol{\mu}\|_2^2 \|\boldsymbol{w}_O\|_2^2} \right) \cdot O\left( \eta \|\boldsymbol{\mu}\|_2^4 \sigma_h^2 d_h^{\frac{3}{4}} \right)$$

$$= O\left( \max\{\|\boldsymbol{\mu}\|_2^2, \sigma_p^2 d\} \cdot \sigma_h^2 \cdot \sqrt{d_h \log(6N^2 M^2/\delta)} \right) + O\left( \|\boldsymbol{\mu}\|_2^2 \sigma_h^2 d_h^{\frac{1}{2}} \right)$$

$$= O\left( \max\{\|\boldsymbol{\mu}\|_2^2, \sigma_p^2 d\} \cdot \sigma_h^2 \cdot \sqrt{d_h \log(6N^2 M^2/\delta)} \right)$$

$$\tag{216}$$

Similarly to $\langle \boldsymbol{q}_+^{(t)}, \boldsymbol{k}_+^{(t)} \rangle$, it is easy to know that the inner product does not change by a magnitude more than the product of $\max\{\alpha, \beta\}$ and $\max\{\langle \boldsymbol{q}, \boldsymbol{q} \rangle, \langle \boldsymbol{k}, \boldsymbol{k} \rangle\}$ in a single iteration, which can be expressed as follows

$$\left| \langle \boldsymbol{q}^{(t+1)}, \boldsymbol{k}^{(t+1)} \rangle - \langle \boldsymbol{q}^{(t)}, \boldsymbol{k}^{(t)} \rangle \right|$$

$$= O\left( \max\{ \frac{\eta \|\boldsymbol{\mu}\|_2^2}{d_h^{\frac{1}{4}}}, \frac{\eta \sigma_p^2 d}{d_h^{\frac{1}{4}} N} \} \right) \cdot \Theta(\max\{\|\boldsymbol{\mu}\|_2^2 \sigma_h^2 d_h, \sigma_p^2 \sigma_h^2 d d_h\})$$

$$= O\left( \frac{\eta \|\boldsymbol{\mu}\|_2^2}{d_h^{\frac{1}{4}}} \right) \cdot \Theta(\max\{\|\boldsymbol{\mu}\|_2^2 \sigma_h^2 d_h, \sigma_p^2 \sigma_h^2 d d_h\})$$

$$= O\left( \eta \|\boldsymbol{\mu}\|_2^2 \sigma_h^2 d_h^{\frac{3}{4}} \cdot \max\{\|\boldsymbol{\mu}\|_2^2, \sigma_p^2 d\} \right)$$

$$\tag{217}$$

where the second equality is by the condition that $N \cdot \text{SNR}^2 = \Omega(1)$.

Taking a summation, we obtain that

$$
\begin{aligned}
\left|\langle \boldsymbol{q}^{(T+1)}, \boldsymbol{k}^{(T+1)} \rangle - \langle \boldsymbol{q}^{(0)}, \boldsymbol{k}^{(0)} \rangle\right| &\leq \sum_{t=0}^{T-1} \left|\langle \boldsymbol{q}^{(t+1)}, \boldsymbol{k}^{(t+1)} \rangle - \langle \boldsymbol{q}^{(t)}, \boldsymbol{k}^{(t)} \rangle\right| \\
&\leq \sum_{t=0}^{T_0-1} O\left(\eta \|\boldsymbol{\mu}\|_2^2 \sigma_h^2 d_h^{\frac{3}{4}} \cdot \max\{\|\boldsymbol{\mu}\|_2^2, \sigma_p^2 d\}\right) \\
&= O\left(\frac{1}{\eta d_h^{\frac{1}{4}} \|\boldsymbol{\mu}\|_2^2 \|\boldsymbol{w}_O\|_2^2}\right) \cdot O\left(\eta \|\boldsymbol{\mu}\|_2^2 \sigma_h^2 d_h^{\frac{3}{4}} \cdot \max\{\|\boldsymbol{\mu}\|_2^2, \sigma_p^2 d\}\right) \\
&= O\left(\max\{\|\boldsymbol{\mu}\|_2^2, \sigma_p^2 d\} \cdot \sigma_h^2 d_h^{\frac{1}{2}}\right).
\end{aligned}
\tag{218}
$$

It is clear that the magnitude of $\langle \boldsymbol{q}^{(T+1)}, \boldsymbol{k}^{(T+1)} \rangle - \langle \boldsymbol{q}^{(0)}, \boldsymbol{k}^{(0)} \rangle$ is smaller than $\max\{\|\boldsymbol{\mu}\|_2^2, \sigma_p^2 d\} \cdot \sigma_h^2 \cdot \sqrt{d_h \log(6N^2 M^2/\delta)}$, Thus the magnitude of the bound for $\langle \boldsymbol{q}^{(T+1)}, \boldsymbol{k}^{(T+1)} \rangle$ is the same as that of $\langle \boldsymbol{q}^{(T)}, \boldsymbol{k}^{(T)} \rangle$. The proof for $\langle \boldsymbol{q}^{(T+1)}, \boldsymbol{q}^{(T+1)} \rangle$ and $\langle \boldsymbol{k}^{(T+1)}, \boldsymbol{k}^{(T+1)} \rangle$ is exactly the same, and we can conclude the proof by an induction.

## F.4 Lower Bounds of $\alpha$ and $\beta$

In this subsection, we present some bounds for $\alpha$ and $\beta$ which can be used in D.2 and D.3. All the calculations in this subsection are based on the precise expression for $\alpha$ and $\beta$ in F.1 and assume that $\mathcal{B}(T_1), \ldots, \mathcal{B}(s), \mathcal{D}(T_1), \ldots, \mathcal{D}(s-1)$ hold ($s \in [T_1, t]$). Then the following propositions hold:

$$
V_+^{(s)} \geq 3M \cdot |V_{n,i}^{(s)}|,
$$

$$
V_-^{(s)} \leq -3M \cdot |V_{n,i}^{(s)}|,
$$

$$
softmax(\langle \boldsymbol{q}_\pm^{(s)}, \boldsymbol{k}_\pm^{(s)} \rangle), softmax(\langle \boldsymbol{q}_{n,i}^{(s)}, \boldsymbol{k}_\pm^{(s)} \rangle) \geq \frac{1}{M} - o(1),
$$

$$
softmax(\langle \boldsymbol{q}_\pm^{(s)}, \boldsymbol{k}_{n,j}^{(s)} \rangle), softmax(\langle \boldsymbol{q}_{n,i}^{(s)}, \boldsymbol{k}_{n,j}^{(s)} \rangle) \leq \frac{1}{M} + o(1).
$$

Now we give the bounds respectively for $\alpha_{+,+}^{(s)}, \alpha_{n,+,i}^{(s)}, \alpha_{-,-}^{(s)}, \alpha_{n,-,i}^{(s)}, \alpha_{n,i,+}^{(s)}, \alpha_{n,i,-}^{(s)}, \alpha_{n,i,n',i'}^{(s)}, \beta_{+,+}^{(s)}, \beta_{n,+,i}^{(s)}, \beta_{-,-}^{(s)}, \beta_{n,-,i}^{(s)}, \beta_{n,i,+}^{(s)}, \beta_{n,i,-}^{(s)}, \beta_{n,i,n',i'}^{(s)}$.

$$\alpha_{+,+}^{(s)} = \frac{\eta}{NM} \sum_{n \in S_+} -\ell_n'^{(s)} \|\boldsymbol{\mu}\|_2^2 \cdot \frac{\exp(\langle \boldsymbol{q}_+^{(s)}, \boldsymbol{k}_+^{(s)} \rangle)}{\exp(\langle \boldsymbol{q}_+^{(s)}, \boldsymbol{k}_+^{(s)} \rangle) + \sum\limits_{j=2}^{M} \exp(\langle \boldsymbol{q}_+^{(s)}, \boldsymbol{k}_{n,j}^{(s)} \rangle)}$$

$$\cdot \left( V_+^{(s)} \left( 1 - \frac{\exp(\langle \boldsymbol{q}_+^{(s)}, \boldsymbol{k}_+^{(s)} \rangle)}{\exp(\langle \boldsymbol{q}_+^{(s)}, \boldsymbol{k}_+^{(s)} \rangle) + \sum\limits_{j=2}^{M} \exp(\langle \boldsymbol{q}_+^{(s)}, \boldsymbol{k}_{n,j}^{(s)} \rangle)} \right) \right.$$

$$\left. - \sum_{i=2}^{M} \left( V_{n,i}^{(s)} \cdot \frac{\exp(\langle \boldsymbol{q}_+^{(s)}, \boldsymbol{k}_{n,i}^{(s)} \rangle)}{\exp(\langle \boldsymbol{q}_+^{(s)}, \boldsymbol{k}_+^{(s)} \rangle) + \sum\limits_{j=2}^{M} \exp(\langle \boldsymbol{q}_+^{(s)}, \boldsymbol{k}_{n,j}^{(s)} \rangle)} \right) \right)$$

$$\geq \frac{\eta}{NM} \sum_{n \in S_+} -\ell_n'^{(s)} \|\boldsymbol{\mu}\|_2^2 \cdot \frac{\exp(\langle \boldsymbol{q}_+^{(s)}, \boldsymbol{k}_+^{(s)} \rangle)}{\exp(\langle \boldsymbol{q}_+^{(s)}, \boldsymbol{k}_+^{(s)} \rangle) + \sum\limits_{j=2}^{M} \exp(\langle \boldsymbol{q}_+^{(s)}, \boldsymbol{k}_{n,j}^{(s)} \rangle)} \tag{219}$$

$$\cdot \left( V_+^{(s)} \left( 1 - \frac{\exp(\langle \boldsymbol{q}_+^{(s)}, \boldsymbol{k}_+^{(s)} \rangle)}{\exp(\langle \boldsymbol{q}_+^{(s)}, \boldsymbol{k}_+^{(s)} \rangle) + \sum\limits_{j=2}^{M} \exp(\langle \boldsymbol{q}_+^{(s)}, \boldsymbol{k}_{n,j}^{(s)} \rangle)} \right) \right.$$

$$\left. - \frac{1}{2} \cdot V_+^{(s)} \sum_{i=2}^{M} \cdot \frac{\exp(\langle \boldsymbol{q}_+^{(s)}, \boldsymbol{k}_{n,i}^{(s)} \rangle)}{\exp(\langle \boldsymbol{q}_+^{(s)}, \boldsymbol{k}_+^{(s)} \rangle) + \sum\limits_{j=2}^{M} \exp(\langle \boldsymbol{q}_+^{(s)}, \boldsymbol{k}_{n,j}^{(s)} \rangle)} \right)$$

$$\geq \frac{\eta}{2NM} \sum_{n \in S_+} -\ell_n'^{(s)} \|\boldsymbol{\mu}\|_2^2 V_+^{(s)} \cdot \frac{\exp(\langle \boldsymbol{q}_+^{(s)}, \boldsymbol{k}_+^{(s)} \rangle)}{\exp(\langle \boldsymbol{q}_+^{(s)}, \boldsymbol{k}_+^{(s)} \rangle) + \sum\limits_{j=2}^{M} \exp(\langle \boldsymbol{q}_+^{(s)}, \boldsymbol{k}_{n,j}^{(s)} \rangle)}$$

$$\cdot \left( 1 - \frac{\exp(\langle \boldsymbol{q}_+^{(s)}, \boldsymbol{k}_+^{(s)} \rangle)}{\exp(\langle \boldsymbol{q}_+^{(s)}, \boldsymbol{k}_+^{(s)} \rangle) + \sum\limits_{j=2}^{M} \exp(\langle \boldsymbol{q}_+^{(s)}, \boldsymbol{k}_{n,j}^{(s)} \rangle)} \right)$$

where the first inequality is by $V_+^{(s)} \geq 3M \cdot |V_{n,i}^{(s)}|$, the second inequality is by the fact that the sum of attention equal to 1. Similarly, we have

$$\beta_{+,+}^{(s)} \geq$$

$$\frac{\eta}{2NM} \sum_{n \in S_+} -\ell_n'^{(s)} \|\boldsymbol{\mu}\|_2^2 V_+^{(s)} \cdot \frac{\exp(\langle \boldsymbol{q}_+^{(s)}, \boldsymbol{k}_+^{(s)} \rangle)}{\exp(\langle \boldsymbol{q}_+^{(s)}, \boldsymbol{k}_+^{(s)} \rangle) + (M-1) \exp(\max\limits_j \{ \langle \boldsymbol{q}_+^{(s)}, \boldsymbol{k}_{n,j}^{(s)} \rangle \})} \tag{220}$$

$$\cdot \frac{\exp(\max\limits_j \{ \langle \boldsymbol{q}_+^{(s)}, \boldsymbol{k}_{n,j}^{(s)} \rangle \})}{\exp(\langle \boldsymbol{q}_+^{(s)}, \boldsymbol{k}_+^{(s)} \rangle) + (M-1) \exp(\max\limits_j \{ \langle \boldsymbol{q}_+^{(s)}, \boldsymbol{k}_{n,j}^{(s)} \rangle \})},$$

$$\alpha_{-,-}^{(s)} \geq$$

$$\frac{\eta}{2NM} \sum_{n \in S_-} \ell_n'^{(s)} \|\boldsymbol{\mu}\|_2^2 V_-^{(s)} \cdot \frac{\exp(\langle \boldsymbol{q}_-^{(s)}, \boldsymbol{k}_-^{(s)} \rangle)}{\exp(\langle \boldsymbol{q}_-^{(s)}, \boldsymbol{k}_-^{(s)} \rangle) + (M-1) \exp(\max\limits_j \{ \langle \boldsymbol{q}_-^{(s)}, \boldsymbol{k}_{n,j}^{(s)} \rangle \})} \tag{221}$$

$$\cdot \frac{\exp(\max\limits_j \{ \langle \boldsymbol{q}_-^{(s)}, \boldsymbol{k}_{n,j}^{(s)} \rangle \})}{\exp(\langle \boldsymbol{q}_-^{(s)}, \boldsymbol{k}_-^{(s)} \rangle) + (M-1) \exp(\max\limits_j \{ \langle \boldsymbol{q}_-^{(s)}, \boldsymbol{k}_{n,j}^{(s)} \rangle \})},$$

$$\beta_{-,-}^{(s)} \geq$$

$$\frac{\eta}{2NM} \sum_{n \in S_-} \ell_n'^{(s)} \|\boldsymbol{\mu}\|_2^2 V_-^{(s)} \cdot \frac{\exp(\langle \boldsymbol{q}_-^{(s)}, \boldsymbol{k}_-^{(s)} \rangle)}{\exp(\langle \boldsymbol{q}_-^{(s)}, \boldsymbol{k}_-^{(s)} \rangle) + (M-1)\exp(\max_j \{\langle \boldsymbol{q}_-^{(s)}, \boldsymbol{k}_{n,j}^{(s)} \rangle\})} \tag{222}$$

$$\cdot \frac{\exp(\max_j \{\langle \boldsymbol{q}_-^{(s)}, \boldsymbol{k}_{n,j}^{(s)} \rangle\})}{\exp(\langle \boldsymbol{q}_-^{(s)}, \boldsymbol{k}_-^{(s)} \rangle) + (M-1)\exp(\max_j \{\langle \boldsymbol{q}_-^{(s)}, \boldsymbol{k}_{n,j}^{(s)} \rangle\})}.$$

$$\alpha_{n,+,i}^{(s)} = -\frac{\eta}{NM} \ell_n'^{(s)} \|\boldsymbol{\mu}\|_2^2 \cdot \frac{\exp(\langle \boldsymbol{q}_+^{(s)}, \boldsymbol{k}_{n,i}^{(s)} \rangle)}{\exp(\langle \boldsymbol{q}_+^{(s)}, \boldsymbol{k}_+^{(s)} \rangle) + \sum\limits_{j=2}^{M} \exp(\langle \boldsymbol{q}_+^{(s)}, \boldsymbol{k}_{n,j}^{(s)} \rangle)}$$

$$\cdot \Big( -V_+^{(s)} \cdot \frac{\exp(\langle \boldsymbol{q}_+^{(s)}, \boldsymbol{k}_+^{(s)} \rangle)}{\exp(\langle \boldsymbol{q}_+^{(s)}, \boldsymbol{k}_+^{(s)} \rangle) + \sum\limits_{j=2}^{M} \exp(\langle \boldsymbol{q}_+^{(s)}, \boldsymbol{k}_{n,j}^{(s)} \rangle)}$$

$$+ V_{n,i}^{(s)} \Big( 1 - \frac{\exp(\langle \boldsymbol{q}_+^{(s)}, \boldsymbol{k}_{n,i}^{(s)} \rangle)}{\exp(\langle \boldsymbol{q}_+^{(s)}, \boldsymbol{k}_+^{(s)} \rangle) + \sum\limits_{j=2}^{M} \exp(\langle \boldsymbol{q}_+^{(s)}, \boldsymbol{k}_{n,j}^{(s)} \rangle)} \Big)$$

$$- \sum_{k \neq i} \Big( V_{n,k}^{(s)} \cdot \frac{\exp(\langle \boldsymbol{q}_+^{(s)}, \boldsymbol{k}_{n,k}^{(s)} \rangle)}{\exp(\langle \boldsymbol{q}_+^{(s)}, \boldsymbol{k}_+^{(s)} \rangle) + \sum\limits_{j=2}^{M} \exp(\langle \boldsymbol{q}_+^{(s)}, \boldsymbol{k}_{n,j}^{(s)} \rangle)} \Big) \Big)$$

$$\leq -\frac{\eta}{NM} \ell_n'^{(s)} \|\boldsymbol{\mu}\|_2^2 \cdot \frac{\exp(\langle \boldsymbol{q}_+^{(s)}, \boldsymbol{k}_{n,i}^{(s)} \rangle)}{\exp(\langle \boldsymbol{q}_+^{(s)}, \boldsymbol{k}_+^{(s)} \rangle) + \sum\limits_{j=2}^{M} \exp(\langle \boldsymbol{q}_+^{(s)}, \boldsymbol{k}_{n,j}^{(s)} \rangle)}$$

$$\cdot \Big( -V_+^{(s)} \cdot \frac{\exp(\langle \boldsymbol{q}_+^{(s)}, \boldsymbol{k}_+^{(s)} \rangle)}{\exp(\langle \boldsymbol{q}_+^{(s)}, \boldsymbol{k}_+^{(s)} \rangle) + \sum\limits_{j=2}^{M} \exp(\langle \boldsymbol{q}_+^{(s)}, \boldsymbol{k}_{n,j}^{(s)} \rangle)}$$

$$+ |V_{n,i}^{(s)}| + \sum_{k \neq i} \Big( |V_{n,k}^{(s)}| \cdot \big( \frac{1}{M} + o(1) \big) \big) \Big)$$

$$\leq -\frac{\eta}{NM} \ell_n'^{(s)} \|\boldsymbol{\mu}\|_2^2 \cdot \frac{\exp(\langle \boldsymbol{q}_+^{(s)}, \boldsymbol{k}_{n,i}^{(s)} \rangle)}{\exp(\langle \boldsymbol{q}_+^{(s)}, \boldsymbol{k}_+^{(s)} \rangle) + \sum\limits_{j=2}^{M} \exp(\langle \boldsymbol{q}_+^{(s)}, \boldsymbol{k}_{n,j}^{(s)} \rangle)}$$

$$\cdot \Big( -V_+^{(s)} \cdot \frac{\exp(\langle \boldsymbol{q}_+^{(s)}, \boldsymbol{k}_+^{(s)} \rangle)}{\exp(\langle \boldsymbol{q}_+^{(s)}, \boldsymbol{k}_+^{(s)} \rangle) + \sum\limits_{j=2}^{M} \exp(\langle \boldsymbol{q}_+^{(s)}, \boldsymbol{k}_{n,j}^{(s)} \rangle)} + 2M \max_l |V_{n,l}^{(s)}| \Big)$$

$$\leq -\frac{\eta}{NM} \ell_n'^{(s)} \|\boldsymbol{\mu}\|_2^2 \cdot \frac{\exp(\langle \boldsymbol{q}_+^{(s)}, \boldsymbol{k}_{n,i}^{(s)} \rangle)}{\exp(\langle \boldsymbol{q}_+^{(s)}, \boldsymbol{k}_+^{(s)} \rangle) + \sum\limits_{j=2}^{M} \exp(\langle \boldsymbol{q}_+^{(s)}, \boldsymbol{k}_{n,j}^{(s)} \rangle)}$$

$$\cdot \Big( -V_+^{(s)} \cdot \frac{\exp(\langle \boldsymbol{q}_+^{(s)}, \boldsymbol{k}_+^{(s)} \rangle)}{\exp(\langle \boldsymbol{q}_+^{(s)}, \boldsymbol{k}_+^{(s)} \rangle) + \sum\limits_{j=2}^{M} \exp(\langle \boldsymbol{q}_+^{(s)}, \boldsymbol{k}_{n,j}^{(s)} \rangle)}$$

$$+ \frac{3}{4} V_+^{(s)} \frac{\exp(\langle \boldsymbol{q}_+^{(s)}, \boldsymbol{k}_+^{(s)}\rangle)}{\exp(\langle \boldsymbol{q}_+^{(s)}, \boldsymbol{k}_+^{(s)}\rangle) + \sum\limits_{j=2}^{M} \exp(\langle \boldsymbol{q}_+^{(s)}, \boldsymbol{k}_{n,j}^{(s)}\rangle)} \Big)$$

$$= \frac{\eta}{4NM} \ell_n'^{(s)} \|\boldsymbol{\mu}\|_2^2 V_+^{(s)} \cdot \frac{\exp(\langle \boldsymbol{q}_+^{(s)}, \boldsymbol{k}_{n,i}^{(s)}\rangle)}{\exp(\langle \boldsymbol{q}_+^{(s)}, \boldsymbol{k}_+^{(s)}\rangle) + \sum\limits_{j=2}^{M} \exp(\langle \boldsymbol{q}_+^{(s)}, \boldsymbol{k}_{n,j}^{(s)}\rangle)}$$

$$\cdot \frac{\exp(\langle \boldsymbol{q}_+^{(s)}, \boldsymbol{k}_+^{(s)}\rangle)}{\exp(\langle \boldsymbol{q}_+^{(s)}, \boldsymbol{k}_+^{(s)}\rangle) + \sum\limits_{j=2}^{M} \exp(\langle \boldsymbol{q}_+^{(s)}, \boldsymbol{k}_{n,j}^{(s)}\rangle)},$$

where the third inequality is by $V_+^{(s)} \geq 3M \cdot |V_{n,i}^{(s)}|$ and $softmax(\langle \boldsymbol{q}_+^{(s)}, \boldsymbol{k}_+^{(s)}\rangle) \geq \frac{1}{M} - o(1)$. Similarly, we have

$$\alpha_{n,-,i}^{(s)} \leq$$

$$- \frac{\eta}{4NM} \ell_n'^{(s)} \|\boldsymbol{\mu}\|_2^2 V_-^{(s)} \cdot \frac{\exp(\langle \boldsymbol{q}_-^{(s)}, \boldsymbol{k}_{n,i}^{(s)}\rangle)}{\exp(\langle \boldsymbol{q}_-^{(s)}, \boldsymbol{k}_-^{(s)}\rangle) + \sum\limits_{j=2}^{M} \exp(\langle \boldsymbol{q}_-^{(s)}, \boldsymbol{k}_{n,j}^{(s)}\rangle)}$$

$$\cdot \frac{\exp(\langle \boldsymbol{q}_-^{(s)}, \boldsymbol{k}_-^{(s)}\rangle)}{\exp(\langle \boldsymbol{q}_-^{(s)}, \boldsymbol{k}_-^{(s)}\rangle) + \sum\limits_{j=2}^{M} \exp(\langle \boldsymbol{q}_-^{(s)}, \boldsymbol{k}_{n,j}^{(s)}\rangle)}.$$

$$\beta_{n,+,i}^{(s)} = -\frac{\eta \|\boldsymbol{\mu}\|_2^2}{NM} \ell_n'^{(s)}$$

$$\cdot \Big( V_+^{(s)} \big( \frac{\exp(\langle \boldsymbol{q}_{n,i}^{(s)}, \boldsymbol{k}_+^{(s)}\rangle)}{\exp(\langle \boldsymbol{q}_{n,i}^{(s)}, \boldsymbol{k}_+^{(s)}\rangle) + \sum\limits_{j=2}^{M} \exp(\langle \boldsymbol{q}_{n,i}^{(s)}, \boldsymbol{k}_{n,j}^{(s)}\rangle)}$$

$$- \big( \frac{\exp(\langle \boldsymbol{q}_{n,i}^{(s)}, \boldsymbol{k}_+^{(s)}\rangle)}{\exp(\langle \boldsymbol{q}_{n,i}^{(s)}, \boldsymbol{k}_+^{(s)}\rangle) + \sum\limits_{j=2}^{M} \exp(\langle \boldsymbol{q}_{n,i}^{(s)}, \boldsymbol{k}_{n,j}^{(s)}\rangle)} \big)^2 \big)$$

$$- \sum\limits_{k=2}^{M} \big( V_{n,i}^{(s)} \cdot \frac{\exp(\langle \boldsymbol{q}_{n,i}^{(s)}, \boldsymbol{k}_+^{(s)}\rangle)}{\exp(\langle \boldsymbol{q}_{n,i}^{(s)}, \boldsymbol{k}_+^{(s)}\rangle) + \sum\limits_{j=2}^{M} \exp(\langle \boldsymbol{q}_{n,i}^{(s)}, \boldsymbol{k}_{n,j}^{(s)}\rangle)}$$

$$\cdot \frac{\exp(\langle \boldsymbol{q}_{n,i}^{(s)}, \boldsymbol{k}_{n,k}^{(s)}\rangle)}{\exp(\langle \boldsymbol{q}_{n,i}^{(s)}, \boldsymbol{k}_+^{(s)}\rangle) + \sum\limits_{j=2}^{M} \exp(\langle \boldsymbol{q}_{n,i}^{(s)}, \boldsymbol{k}_{n,j}^{(s)}\rangle)} \big) \Big)$$

$$= -\frac{\eta \|\boldsymbol{\mu}\|_2^2}{NM} \ell_n'^{(s)} \frac{\exp(\langle \boldsymbol{q}_{n,i}^{(s)}, \boldsymbol{k}_+^{(s)}\rangle)}{\exp(\langle \boldsymbol{q}_{n,i}^{(s)}, \boldsymbol{k}_+^{(s)}\rangle) + \sum\limits_{j=2}^{M} \exp(\langle \boldsymbol{q}_{n,i}^{(s)}, \boldsymbol{k}_{n,j}^{(s)}\rangle)}$$

$$\cdot \Big( V_+^{(s)} \big( 1 - \frac{\exp(\langle \boldsymbol{q}_{n,i}^{(s)}, \boldsymbol{k}_+^{(s)}\rangle)}{\exp(\langle \boldsymbol{q}_{n,i}^{(s)}, \boldsymbol{k}_+^{(s)}\rangle) + \sum\limits_{j=2}^{M} \exp(\langle \boldsymbol{q}_{n,i}^{(s)}, \boldsymbol{k}_{n,j}^{(s)}\rangle)} \big)$$

$$- \sum\limits_{k=2}^{M} \big( V_{n,i}^{(s)} \cdot \frac{\exp(\langle \boldsymbol{q}_{n,i}^{(s)}, \boldsymbol{k}_{n,k}^{(s)}\rangle)}{\exp(\langle \boldsymbol{q}_{n,i}^{(s)}, \boldsymbol{k}_+^{(s)}\rangle) + \sum\limits_{j=2}^{M} \exp(\langle \boldsymbol{q}_{n,i}^{(s)}, \boldsymbol{k}_{n,j}^{(s)}\rangle)} \big) \Big)$$

$$\geq -\frac{\eta\|\boldsymbol{\mu}\|_2^2}{NM}\ell_n'^{(s)}\frac{\exp(\langle\boldsymbol{q}_{n,i}^{(s)},\boldsymbol{k}_+^{(s)}\rangle)}{\exp(\langle\boldsymbol{q}_{n,i}^{(s)},\boldsymbol{k}_+^{(s)}\rangle)+\sum\limits_{j=2}^{M}\exp(\langle\boldsymbol{q}_{n,i}^{(s)},\boldsymbol{k}_{n,j}^{(s)}\rangle)}$$

$$\cdot\Big(V_+^{(s)}\big(1-\frac{\exp(\langle\boldsymbol{q}_{n,i}^{(s)},\boldsymbol{k}_+^{(s)}\rangle)}{\exp(\langle\boldsymbol{q}_{n,i}^{(s)},\boldsymbol{k}_+^{(s)}\rangle)+\sum\limits_{j=2}^{M}\exp(\langle\boldsymbol{q}_{n,i}^{(s)},\boldsymbol{k}_{n,j}^{(s)}\rangle)}\big)$$

$$-\frac{1}{2}V_+^{(s)}\sum\limits_{k=2}^{M}\cdot\frac{\exp(\langle\boldsymbol{q}_{n,i}^{(s)},\boldsymbol{k}_{n,k}^{(s)}\rangle)}{\exp(\langle\boldsymbol{q}_{n,i}^{(s)},\boldsymbol{k}_+^{(s)}\rangle)+\sum\limits_{j=2}^{M}\exp(\langle\boldsymbol{q}_{n,i}^{(s)},\boldsymbol{k}_{n,j}^{(s)}\rangle)}\big)$$

$$\geq -\frac{\eta\|\boldsymbol{\mu}\|_2^2}{2NM}\ell_n'^{(s)}V_+^{(s)}\frac{\exp(\langle\boldsymbol{q}_{n,i}^{(s)},\boldsymbol{k}_+^{(s)}\rangle)}{\exp(\langle\boldsymbol{q}_{n,i}^{(s)},\boldsymbol{k}_+^{(s)}\rangle)+\sum\limits_{j=2}^{M}\exp(\langle\boldsymbol{q}_{n,i}^{(s)},\boldsymbol{k}_{n,j}^{(s)}\rangle)}$$

$$\cdot\big(1-\frac{\exp(\langle\boldsymbol{q}_{n,i}^{(s)},\boldsymbol{k}_+^{(s)}\rangle)}{\exp(\langle\boldsymbol{q}_{n,i}^{(s)},\boldsymbol{k}_+^{(s)}\rangle)+\sum\limits_{j=2}^{M}\exp(\langle\boldsymbol{q}_{n,i}^{(s)},\boldsymbol{k}_{n,j}^{(s)}\rangle)}\big),$$

where the first inequality is by $V_+^{(s)}\geq 3M\cdot|V_{n,i}^{(s)}|$, the second inequality is by $\sum\limits_{k=2}^{M}softmax(\langle\boldsymbol{q}_{n,i}^{(s)},\boldsymbol{k}_{n,k}^{(s)}\rangle)=\big(1-softmax(\langle\boldsymbol{q}_{n,i}^{(s)},\boldsymbol{k}_+^{(s)}\rangle)\big)$. Similarly, we have

$$\beta_{n,-,i}^{(s)}\geq$$

$$\frac{\eta\|\boldsymbol{\mu}\|_2^2}{2NM}\ell_n'^{(s)}V_-^{(s)}\frac{\exp(\langle\boldsymbol{q}_{n,i}^{(s)},\boldsymbol{k}_-^{(s)}\rangle)}{\exp(\langle\boldsymbol{q}_{n,i}^{(s)},\boldsymbol{k}_-^{(s)}\rangle)+\sum\limits_{j=2}^{M}\exp(\langle\boldsymbol{q}_{n,i}^{(s)},\boldsymbol{k}_{n,j}^{(s)}\rangle)}$$

$$\cdot\big(1-\frac{\exp(\langle\boldsymbol{q}_{n,i}^{(s)},\boldsymbol{k}_-^{(s)}\rangle)}{\exp(\langle\boldsymbol{q}_{n,i}^{(s)},\boldsymbol{k}_-^{(s)}\rangle)+\sum\limits_{j=2}^{M}\exp(\langle\boldsymbol{q}_{n,i}^{(s)},\boldsymbol{k}_{n,j}^{(s)}\rangle)}\big).$$

$$\alpha_{n',i',+}^{(s)}=\frac{\eta\|\boldsymbol{\xi}_{n',i'}\|_2^2}{NM}-\ell_{n'}'(\theta(s))$$

$$\cdot\Big(V_+^{(s)}\big(\frac{\exp(\langle\boldsymbol{q}_{n',i'}^{(s)},\boldsymbol{k}_+^{(s)}\rangle)}{\exp(\langle\boldsymbol{q}_{n',i'}^{(s)},\boldsymbol{k}_+^{(s)}\rangle)+\sum\limits_{j=2}^{M}\exp(\langle\boldsymbol{q}_{n',i'}^{(s)},\boldsymbol{k}_{n',j}^{(s)}\rangle)}$$

$$-\big(\frac{\exp(\langle\boldsymbol{q}_{n',i'}^{(s)},\boldsymbol{k}_+^{(s)}\rangle)}{\exp(\langle\boldsymbol{q}_{n',i'}^{(s)},\boldsymbol{k}_+^{(s)}\rangle)+\sum\limits_{j=2}^{M}\exp(\langle\boldsymbol{q}_{n',i'}^{(s)},\boldsymbol{k}_{n',j}^{(s)}\rangle)}\big)^2\big)$$

$$-\sum\limits_{k=2}^{M}\big(V_{n',i'}^{(s)}\cdot\frac{\exp(\langle\boldsymbol{q}_{n',i'}^{(s)},\boldsymbol{k}_+^{(s)}\rangle)}{\exp(\langle\boldsymbol{q}_{n',i'}^{(s)},\boldsymbol{k}_+^{(s)}\rangle)+\sum\limits_{j=2}^{M}\exp(\langle\boldsymbol{q}_{n',i'}^{(s)},\boldsymbol{k}_{n',j}^{(s)}\rangle)}$$

$$\cdot\frac{\exp(\langle\boldsymbol{q}_{n',i'}^{(s)},\boldsymbol{k}_{n',k}^{(s)}\rangle)}{\exp(\langle\boldsymbol{q}_{n',i'}^{(s)},\boldsymbol{k}_+^{(s)}\rangle)+\sum\limits_{j=2}^{M}\exp(\langle\boldsymbol{q}_{n',i'}^{(s)},\boldsymbol{k}_{n',j}^{(s)}\rangle)}\big)\Big)$$

$$+\{lower\ order\ term\}$$

$$= \frac{\eta\|\boldsymbol{\xi}_{n',i'}\|_2^2}{NM} - \ell_{n'}'(\theta(s)) \frac{\exp(\langle \boldsymbol{q}_{n',i'}^{(s)}, \boldsymbol{k}_+^{(s)}\rangle)}{\exp(\langle \boldsymbol{q}_{n',i'}^{(s)}, \boldsymbol{k}_+^{(s)}\rangle) + \sum\limits_{j=2}^{M} \exp(\langle \boldsymbol{q}_{n',i'}^{(s)}, \boldsymbol{k}_{n',j}^{(s)}\rangle)}$$

$$\cdot \left( V_+^{(s)} \left(1 - \frac{\exp(\langle \boldsymbol{q}_{n',i'}^{(s)}, \boldsymbol{k}_+^{(s)}\rangle)}{\exp(\langle \boldsymbol{q}_{n',i'}^{(s)}, \boldsymbol{k}_+^{(s)}\rangle) + \sum\limits_{j=2}^{M} \exp(\langle \boldsymbol{q}_{n',i'}^{(s)}, \boldsymbol{k}_{n',j}^{(s)}\rangle)} \right) \right.$$

$$\left. - \sum_{k=2}^{M} \left( V_{n',i'}^{(s)} \cdot \frac{\exp(\langle \boldsymbol{q}_{n',i'}^{(s)}, \boldsymbol{k}_{n',k}^{(s)}\rangle)}{\exp(\langle \boldsymbol{q}_{n',i'}^{(s)}, \boldsymbol{k}_+^{(s)}\rangle) + \sum\limits_{j=2}^{M} \exp(\langle \boldsymbol{q}_{n',i'}^{(s)}, \boldsymbol{k}_{n',j}^{(s)}\rangle)} \right) \right)$$

$$+ \{lower\ order\ term\}$$

$$\geq \frac{\eta\|\boldsymbol{\xi}_{n',i'}\|_2^2}{NM} - \ell_{n'}'(\theta(s)) \frac{\exp(\langle \boldsymbol{q}_{n',i'}^{(s)}, \boldsymbol{k}_+^{(s)}\rangle)}{\exp(\langle \boldsymbol{q}_{n',i'}^{(s)}, \boldsymbol{k}_+^{(s)}\rangle) + \sum\limits_{j=2}^{M} \exp(\langle \boldsymbol{q}_{n',i'}^{(s)}, \boldsymbol{k}_{n,j}^{(s)}\rangle)}$$

$$\cdot \left( V_+^{(s)} \left(1 - \frac{\exp(\langle \boldsymbol{q}_{n',i'}^{(s)}, \boldsymbol{k}_+^{(s)}\rangle)}{\exp(\langle \boldsymbol{q}_{n',i'}^{(s)}, \boldsymbol{k}_+^{(s)}\rangle) + \sum\limits_{j=2}^{M} \exp(\langle \boldsymbol{q}_{n',i'}^{(s)}, \boldsymbol{k}_{n',j}^{(s)}\rangle)} \right) \right.$$

$$\left. - \frac{1}{2} V_+^{(s)} \sum_{k=2}^{M} \left( \frac{\exp(\langle \boldsymbol{q}_{n',i'}^{(s)}, \boldsymbol{k}_{n',k}^{(s)}\rangle)}{\exp(\langle \boldsymbol{q}_{n',i'}^{(s)}, \boldsymbol{k}_+^{(s)}\rangle) + \sum\limits_{j=2}^{M} \exp(\langle \boldsymbol{q}_{n',i'}^{(s)}, \boldsymbol{k}_{n',j}^{(s)}\rangle)} \right) \right)$$

$$+ \{lower\ order\ term\}$$

$$= \frac{\eta\|\boldsymbol{\xi}_{n',i'}\|_2^2}{2NM} - \ell_{n'}'(\theta(s)) V_+^{(s)} \frac{\exp(\langle \boldsymbol{q}_{n',i'}^{(s)}, \boldsymbol{k}_+^{(s)}\rangle)}{\exp(\langle \boldsymbol{q}_{n',i'}^{(s)}, \boldsymbol{k}_+^{(s)}\rangle) + \sum\limits_{j=2}^{M} \exp(\langle \boldsymbol{q}_{n',i'}^{(s)}, \boldsymbol{k}_{n',j}^{(s)}\rangle)}$$

$$\cdot \left(1 - \frac{\exp(\langle \boldsymbol{q}_{n',i'}^{(s)}, \boldsymbol{k}_+^{(s)}\rangle)}{\exp(\langle \boldsymbol{q}_{n',i'}^{(s)}, \boldsymbol{k}_+^{(s)}\rangle) + \sum\limits_{j=2}^{M} \exp(\langle \boldsymbol{q}_{n',i'}^{(s)}, \boldsymbol{k}_{n',j}^{(s)}\rangle)} \right)$$

$$+ \{lower\ order\ term\}$$

$$\geq \frac{\eta\sigma_p^2 d}{5NM} - \ell_{n'}'(\theta(s)) V_+^{(s)} \frac{\exp(\langle \boldsymbol{q}_{n',i'}^{(s)}, \boldsymbol{k}_+^{(s)}\rangle)}{\exp(\langle \boldsymbol{q}_{n',i'}^{(s)}, \boldsymbol{k}_+^{(s)}\rangle) + \sum\limits_{j=2}^{M} \exp(\langle \boldsymbol{q}_{n',i'}^{(s)}, \boldsymbol{k}_{n',j}^{(s)}\rangle)}$$

$$\cdot \left(1 - \frac{\exp(\langle \boldsymbol{q}_{n',i'}^{(s)}, \boldsymbol{k}_+^{(s)}\rangle)}{\exp(\langle \boldsymbol{q}_{n',i'}^{(s)}, \boldsymbol{k}_+^{(s)}\rangle) + \sum\limits_{j=2}^{M} \exp(\langle \boldsymbol{q}_{n',i'}^{(s)}, \boldsymbol{k}_{n',j}^{(s)}\rangle)} \right),$$

where the $\{lower\ order\ term\}$ is by the property that $\|\boldsymbol{\xi}_{n',i'}\|_2^2$ is much larger than $\langle \boldsymbol{\xi}_{n',i'}, \boldsymbol{\xi}_{n,i}\rangle$ in Lemma C.4 and the condition $d = \widetilde{\Omega}\left(\epsilon^{-2} N^2 d_h\right)$, the first inequality is by $V_+^{(s)} \geq 3M \cdot |V_{n,i}^{(s)}|$, the last inequality is by $\|\boldsymbol{\xi}_{n',i'}\|_2^2 \geq \frac{\sigma_p^2 d}{2}$ and we absorb the $\{lower\ order\ term\}$. Similarly, we have

$$\alpha_{n',i',-}^{(s)} \geq$$

$$\frac{\eta\sigma_p^2 d}{5NM} \ell_{n'}'(\theta(s)) V_-^{(s)} \frac{\exp(\langle \boldsymbol{q}_{n',i'}^{(s)}, \boldsymbol{k}_-^{(s)}\rangle)}{\exp(\langle \boldsymbol{q}_{n',i'}^{(s)}, \boldsymbol{k}_-^{(s)}\rangle) + \sum\limits_{j=2}^{M} \exp(\langle \boldsymbol{q}_{n',i'}^{(s)}, \boldsymbol{k}_{n',j}^{(s)}\rangle)}$$

$$\cdot \Big(1 - \frac{\exp(\langle \boldsymbol{q}_{n',i'}^{(s)}, \boldsymbol{k}_{-}^{(s)}\rangle)}{\exp(\langle \boldsymbol{q}_{n',i'}^{(s)}, \boldsymbol{k}_{-}^{(s)}\rangle) + \sum\limits_{j=2}^{M} \exp(\langle \boldsymbol{q}_{n',i'}^{(s)}, \boldsymbol{k}_{n',j}^{(s)}\rangle)}\Big).$$

$$\beta_{n',i',+}^{(s)} = \frac{\eta \|\boldsymbol{\xi}_{n',i'}\|_2^2}{NM} - \ell_{n'}'(\theta(s))$$

$$\Big(- V_+^{(s)} \Big(\frac{\exp(\langle \boldsymbol{q}_+^{(s)}, \boldsymbol{k}_+^{(s)}\rangle)}{\exp(\langle \boldsymbol{q}_+^{(s)}, \boldsymbol{k}_+^{(s)}\rangle) + \sum\limits_{j=2}^{M} \exp(\langle \boldsymbol{q}_+^{(s)}, \boldsymbol{k}_{n',j}^{(s)}\rangle)}$$

$$\cdot \frac{\exp(\langle \boldsymbol{q}_+^{(s)}, \boldsymbol{k}_{n',i'}^{(s)}\rangle)}{\exp(\langle \boldsymbol{q}_+^{(s)}, \boldsymbol{k}_+^{(s)}\rangle) + \sum\limits_{j=2}^{M} \exp(\langle \boldsymbol{q}_+^{(s)}, \boldsymbol{k}_{n',j}^{(s)}\rangle)}\Big)$$

$$+ V_{n',i'}^{(s)} \Big(\frac{\exp(\langle \boldsymbol{q}_+^{(s)}, \boldsymbol{k}_{n',i'}^{(s)}\rangle)}{\exp(\langle \boldsymbol{q}_+^{(s)}, \boldsymbol{k}_+^{(s)}\rangle) + \sum\limits_{j=2}^{M} \exp(\langle \boldsymbol{q}_+^{(s)}, \boldsymbol{k}_{n',j}^{(s)}\rangle)}$$

$$\cdot \Big(1 - \frac{\exp(\langle \boldsymbol{q}_+^{(s)}, \boldsymbol{k}_{n',i'}^{(s)}\rangle)}{\exp(\langle \boldsymbol{q}_+^{(s)}, \boldsymbol{k}_+^{(s)}\rangle) + \sum\limits_{j=2}^{M} \exp(\langle \boldsymbol{q}_+^{(s)}, \boldsymbol{k}_{n',j}^{(s)}\rangle)}\Big)\Big)\Big)$$

$$+ \{lower\ order\ term\}$$

$$= \frac{\eta \|\boldsymbol{\xi}_{n',i'}\|_2^2}{NM} - \ell_{n'}'(\theta(s)) \cdot \frac{\exp(\langle \boldsymbol{q}_+^{(s)}, \boldsymbol{k}_{n',i'}^{(s)}\rangle)}{\exp(\langle \boldsymbol{q}_+^{(s)}, \boldsymbol{k}_+^{(s)}\rangle) + \sum\limits_{j=2}^{M} \exp(\langle \boldsymbol{q}_+^{(s)}, \boldsymbol{k}_{n',j}^{(s)}\rangle)}$$

$$\Big(- V_+^{(s)} \cdot \frac{\exp(\langle \boldsymbol{q}_+^{(s)}, \boldsymbol{k}_+^{(s)}\rangle)}{\exp(\langle \boldsymbol{q}_+^{(s)}, \boldsymbol{k}_+^{(s)}\rangle) + \sum\limits_{j=2}^{M} \exp(\langle \boldsymbol{q}_+^{(s)}, \boldsymbol{k}_{n',j}^{(s)}\rangle)}$$

$$+ V_{n',i'}^{(s)} \cdot \Big(1 - \frac{\exp(\langle \boldsymbol{q}_+^{(s)}, \boldsymbol{k}_{n',i'}^{(s)}\rangle)}{\exp(\langle \boldsymbol{q}_+^{(s)}, \boldsymbol{k}_+^{(s)}\rangle) + \sum\limits_{j=2}^{M} \exp(\langle \boldsymbol{q}_+^{(s)}, \boldsymbol{k}_{n',j}^{(s)}\rangle)}\Big)\Big)$$

$$+ \{lower\ order\ term\}$$

$$\leq \frac{\eta \|\boldsymbol{\xi}_{n',i'}\|_2^2}{NM} - \ell_{n'}'(\theta(s)) \cdot \frac{\exp(\langle \boldsymbol{q}_+^{(s)}, \boldsymbol{k}_{n',i'}^{(s)}\rangle)}{\exp(\langle \boldsymbol{q}_+^{(s)}, \boldsymbol{k}_+^{(s)}\rangle) + \sum\limits_{j=2}^{M} \exp(\langle \boldsymbol{q}_+^{(s)}, \boldsymbol{k}_{n',j}^{(s)}\rangle)}$$

$$\Big(- V_+^{(s)} \cdot \frac{\exp(\langle \boldsymbol{q}_+^{(s)}, \boldsymbol{k}_+^{(s)}\rangle)}{\exp(\langle \boldsymbol{q}_+^{(s)}, \boldsymbol{k}_+^{(s)}\rangle) + \sum\limits_{j=2}^{M} \exp(\langle \boldsymbol{q}_+^{(s)}, \boldsymbol{k}_{n',j}^{(s)}\rangle)}$$

$$+ \frac{1}{2} V_+^{(s)} \cdot \frac{\exp(\langle \boldsymbol{q}_+^{(s)}, \boldsymbol{k}_+^{(s)}\rangle)}{\exp(\langle \boldsymbol{q}_+^{(s)}, \boldsymbol{k}_+^{(s)}\rangle) + \sum\limits_{j=2}^{M} \exp(\langle \boldsymbol{q}_+^{(s)}, \boldsymbol{k}_{n',j}^{(s)}\rangle)}\Big)$$

$$\leq \frac{\eta \|\boldsymbol{\xi}_{n',i'}\|_2^2}{2NM} \ell_{n'}'(\theta(s)) V_+^{(s)} \cdot \frac{\exp(\langle \boldsymbol{q}_+^{(s)}, \boldsymbol{k}_{n',i'}^{(s)}\rangle)}{\exp(\langle \boldsymbol{q}_+^{(s)}, \boldsymbol{k}_+^{(s)}\rangle) + \sum\limits_{j=2}^{M} \exp(\langle \boldsymbol{q}_+^{(s)}, \boldsymbol{k}_{n',j}^{(s)}\rangle)}$$

$$\cdot \frac{\exp(\langle \boldsymbol{q}_+^{(s)}, \boldsymbol{k}_+^{(s)}\rangle)}{\exp(\langle \boldsymbol{q}_+^{(s)}, \boldsymbol{k}_+^{(s)}\rangle) + \sum\limits_{j=2}^{M} \exp(\langle \boldsymbol{q}_+^{(s)}, \boldsymbol{k}_{n',j}^{(s)}\rangle)}$$

$$+ \{lower\ order\ term\}$$

$$\leq \frac{\eta \sigma_p^2 d}{5NM} \ell'_{n'}(\theta(s)) V_+^{(s)} \cdot \frac{\exp(\langle \boldsymbol{q}_+^{(s)}, \boldsymbol{k}_{n',i'}^{(s)}\rangle)}{\exp(\langle \boldsymbol{q}_+^{(s)}, \boldsymbol{k}_+^{(s)}\rangle) + \sum\limits_{j=2}^{M} \exp(\langle \boldsymbol{q}_+^{(s)}, \boldsymbol{k}_{n',j}^{(s)}\rangle)}$$

$$\cdot \frac{\exp(\langle \boldsymbol{q}_+^{(s)}, \boldsymbol{k}_+^{(s)}\rangle)}{\exp(\langle \boldsymbol{q}_+^{(s)}, \boldsymbol{k}_+^{(s)}\rangle) + \sum\limits_{j=2}^{M} \exp(\langle \boldsymbol{q}_+^{(s)}, \boldsymbol{k}_{n',j}^{(s)}\rangle)},$$

where the $\{lower\ order\ term\}$ is by the property that $\|\boldsymbol{\xi}_{n',i'}\|_2^2$ is much larger than $\langle \boldsymbol{\xi}_{n',i'}, \boldsymbol{\xi}_{n,i}\rangle$ in Lemma C.4 and the condition $d = \widetilde{\Omega}\left(\epsilon^{-2} N^2 d_h\right)$, the first inequality is by $V_+^{(s)} \geq 3M \cdot |V_{n,i}^{(s)}|$, the last inequality is by $\|\boldsymbol{\xi}_{n',i'}\|_2^2 \geq \frac{\sigma_p^2 d}{2}$ and we absorb the $\{lower\ order\ term\}$. Similarly, we have

$$\beta_{n',i',-}^{(s)} \leq$$

$$- \frac{\eta \sigma_p^2 d}{5NM} \ell'_{n'}(\theta(s)) V_-^{(s)} \cdot \frac{\exp(\langle \boldsymbol{q}_-^{(s)}, \boldsymbol{k}_{n',i'}^{(s)}\rangle)}{\exp(\langle \boldsymbol{q}_-^{(s)}, \boldsymbol{k}_-^{(s)}\rangle) + \sum\limits_{j=2}^{M} \exp(\langle \boldsymbol{q}_-^{(s)}, \boldsymbol{k}_{n',j}^{(s)}\rangle)}$$

$$\cdot \frac{\exp(\langle \boldsymbol{q}_-^{(s)}, \boldsymbol{k}_-^{(s)}\rangle)}{\exp(\langle \boldsymbol{q}_-^{(s)}, \boldsymbol{k}_-^{(s)}\rangle) + \sum\limits_{j=2}^{M} \exp(\langle \boldsymbol{q}_-^{(s)}, \boldsymbol{k}_{n',j}^{(s)}\rangle)}.$$

$$\alpha_{n,i,n,j}^{(s)} = \frac{\eta}{NM} - \ell_n'^{(s)} \sum_{k=2}^{M} \langle \boldsymbol{\xi}_{n,i}, \boldsymbol{\xi}_{n,k}\rangle \cdot \frac{\exp(\langle \boldsymbol{q}_{n,k}^{(s)}, \boldsymbol{k}_{n,j}^{(s)}\rangle)}{\exp(\langle \boldsymbol{q}_{n,k}^{(s)}, \boldsymbol{k}_+^{(s)}\rangle) + \sum\limits_{l'=2}^{M} \exp(\langle \boldsymbol{q}_{n,k}^{(s)}, \boldsymbol{k}_{n,l'}^{(s)}\rangle)}$$

$$\cdot \left( - V_+^{(s)} \cdot \frac{\exp(\langle \boldsymbol{q}_{n,k}^{(s)}, \boldsymbol{k}_+^{(s)}\rangle)}{\exp(\langle \boldsymbol{q}_{n,k}^{(s)}, \boldsymbol{k}_+^{(s)}\rangle) + \sum\limits_{l'=2}^{M} \exp(\langle \boldsymbol{q}_{n,k}^{(s)}, \boldsymbol{k}_{n,l'}^{(s)}\rangle)} \right.$$

$$+ V_{n,j}^{(s)} \left(1 - \frac{\exp(\langle \boldsymbol{q}_{n,k}^{(s)}, \boldsymbol{k}_{n,j}^{(s)}\rangle)}{\exp(\langle \boldsymbol{q}_{n,k}^{(s)}, \boldsymbol{k}_+^{(s)}\rangle) + \sum\limits_{l'=2}^{M} \exp(\langle \boldsymbol{q}_{n,k}^{(s)}, \boldsymbol{k}_{n,l'}^{(s)}\rangle)}\right)$$

$$- \sum_{l \neq i} \left( V_{n,l}^{(s)} \cdot \frac{\exp(\langle \boldsymbol{q}_{n,k}^{(s)}, \boldsymbol{k}_{n,l}^{(s)}\rangle)}{\exp(\langle \boldsymbol{q}_{n,k}^{(s)}, \boldsymbol{k}_+^{(s)}\rangle) + \sum\limits_{l'=2}^{M} \exp(\langle \boldsymbol{q}_{n,k}^{(s)}, \boldsymbol{k}_{n,l'}^{(s)}\rangle)} \right) \Bigg)$$

$$= - \frac{\eta \|\boldsymbol{\xi}_{n,i}\|_2^2}{NM} \ell_n'^{(s)} \cdot \frac{\exp(\langle \boldsymbol{q}_{n,i}^{(s)}, \boldsymbol{k}_{n,j}^{(s)}\rangle)}{\exp(\langle \boldsymbol{q}_{n,i}^{(s)}, \boldsymbol{k}_+^{(s)}\rangle) + \sum\limits_{l'=2}^{M} \exp(\langle \boldsymbol{q}_{n,i}^{(s)}, \boldsymbol{k}_{n,l'}^{(s)}\rangle)}$$

$$\cdot \left( - V_+^{(s)} \cdot \frac{\exp(\langle \boldsymbol{q}_{n,i}^{(s)}, \boldsymbol{k}_+^{(s)}\rangle)}{\exp(\langle \boldsymbol{q}_{n,i}^{(s)}, \boldsymbol{k}_+^{(s)}\rangle) + \sum\limits_{l'=2}^{M} \exp(\langle \boldsymbol{q}_{n,i}^{(s)}, \boldsymbol{k}_{n,l'}^{(s)}\rangle)} \right.$$

$$+ V_{n,j}^{(s)} \left(1 - \frac{\exp(\langle \boldsymbol{q}_{n,i}^{(s)}, \boldsymbol{k}_{n,j}^{(s)}\rangle)}{\exp(\langle \boldsymbol{q}_{n,i}^{(s)}, \boldsymbol{k}_+^{(s)}\rangle) + \sum\limits_{l'=2}^{M} \exp(\langle \boldsymbol{q}_{n,i}^{(s)}, \boldsymbol{k}_{n,l'}^{(s)}\rangle)}\right)$$

$$
- \sum_{l \neq i} \left( V_{n,l}^{(s)} \cdot \frac{\exp(\langle \boldsymbol{q}_{n,i}^{(s)}, \boldsymbol{k}_{n,l}^{(s)} \rangle)}{\exp(\langle \boldsymbol{q}_{n,i}^{(s)}, \boldsymbol{k}_+^{(s)} \rangle) + \sum_{l'=2}^{M} \exp(\langle \boldsymbol{q}_{n,i}^{(s)}, \boldsymbol{k}_{n,l'}^{(s)} \rangle)} \right) \Big)
$$

$+ \{lower\ order\ term\}$

$$
\leq -\frac{\eta \|\boldsymbol{\xi}_{n,i}\|_2^2}{NM} \ell_n'^{(s)} \cdot \frac{\exp(\langle \boldsymbol{q}_{n,i}^{(s)}, \boldsymbol{k}_{n,j}^{(s)} \rangle)}{\exp(\langle \boldsymbol{q}_{n,i}^{(s)}, \boldsymbol{k}_+^{(s)} \rangle) + \sum_{l'=2}^{M} \exp(\langle \boldsymbol{q}_{n,i}^{(s)}, \boldsymbol{k}_{n,l'}^{(s)} \rangle)}
$$

$$
\cdot \left( - V_+^{(s)} \cdot \frac{\exp(\langle \boldsymbol{q}_{n,i}^{(s)}, \boldsymbol{k}_+^{(s)} \rangle)}{\exp(\langle \boldsymbol{q}_{n,i}^{(s)}, \boldsymbol{k}_+^{(s)} \rangle) + \sum_{l'=2}^{M} \exp(\langle \boldsymbol{q}_{n,i}^{(s)}, \boldsymbol{k}_{n,l'}^{(s)} \rangle)} \right.
$$

$$
+ |V_{n,j}^{(s)}| + \sum_{l \neq i} \left( |V_{n,l}^{(s)}| \cdot \left( \frac{1}{M} + o(1) \right) \right) \Big)
$$

$+ \{lower\ order\ term\}$

$$
\leq -\frac{\eta \|\boldsymbol{\xi}_{n,i}\|_2^2}{NM} \ell_n'^{(s)} \cdot \frac{\exp(\langle \boldsymbol{q}_{n,i}^{(s)}, \boldsymbol{k}_{n,j}^{(s)} \rangle)}{\exp(\langle \boldsymbol{q}_{n,i}^{(s)}, \boldsymbol{k}_+^{(s)} \rangle) + \sum_{l'=2}^{M} \exp(\langle \boldsymbol{q}_{n,i}^{(s)}, \boldsymbol{k}_{n,l'}^{(s)} \rangle)}
$$

$$
\cdot \left( - V_+^{(s)} \cdot \frac{\exp(\langle \boldsymbol{q}_{n,i}^{(s)}, \boldsymbol{k}_+^{(s)} \rangle)}{\exp(\langle \boldsymbol{q}_{n,i}^{(s)}, \boldsymbol{k}_+^{(s)} \rangle) + \sum_{l'=2}^{M} \exp(\langle \boldsymbol{q}_{n,i}^{(s)}, \boldsymbol{k}_{n,l'}^{(s)} \rangle)} + 2 \max_l |V_{n,l}^{(s)}| \right)
$$

$+ \{lower\ order\ term\}$

$$
\leq -\frac{\eta \|\boldsymbol{\xi}_{n,i}\|_2^2}{NM} \ell_n'^{(s)} \cdot \frac{\exp(\langle \boldsymbol{q}_{n,i}^{(s)}, \boldsymbol{k}_{n,j}^{(s)} \rangle)}{\exp(\langle \boldsymbol{q}_{n,i}^{(s)}, \boldsymbol{k}_+^{(s)} \rangle) + \sum_{l'=2}^{M} \exp(\langle \boldsymbol{q}_{n,i}^{(s)}, \boldsymbol{k}_{n,l'}^{(s)} \rangle)}
$$

$$
\cdot \left( - V_+^{(s)} \cdot \frac{\exp(\langle \boldsymbol{q}_{n,i}^{(s)}, \boldsymbol{k}_+^{(s)} \rangle)}{\exp(\langle \boldsymbol{q}_{n,i}^{(s)}, \boldsymbol{k}_+^{(s)} \rangle) + \sum_{l'=2}^{M} \exp(\langle \boldsymbol{q}_{n,i}^{(s)}, \boldsymbol{k}_{n,l'}^{(s)} \rangle)} \right.
$$

$$
+ \frac{3}{4} V_+^{(s)} \cdot \frac{\exp(\langle \boldsymbol{q}_{n,i}^{(s)}, \boldsymbol{k}_+^{(s)} \rangle)}{\exp(\langle \boldsymbol{q}_{n,i}^{(s)}, \boldsymbol{k}_+^{(s)} \rangle) + \sum_{l'=2}^{M} \exp(\langle \boldsymbol{q}_{n,i}^{(s)}, \boldsymbol{k}_{n,l'}^{(s)} \rangle)} \Big)
$$

$+ \{lower\ order\ term\}$

$$
\leq \frac{\eta \|\boldsymbol{\xi}_{n,i}\|_2^2}{4NM} \ell_n'^{(s)} V_+^{(s)} \cdot \frac{\exp(\langle \boldsymbol{q}_{n,i}^{(s)}, \boldsymbol{k}_{n,j}^{(s)} \rangle)}{\exp(\langle \boldsymbol{q}_{n,i}^{(s)}, \boldsymbol{k}_+^{(s)} \rangle) + \sum_{l'=2}^{M} \exp(\langle \boldsymbol{q}_{n,i}^{(s)}, \boldsymbol{k}_{n,l'}^{(s)} \rangle)}
$$

$$
\cdot \frac{\exp(\langle \boldsymbol{q}_{n,i}^{(s)}, \boldsymbol{k}_+^{(s)} \rangle)}{\exp(\langle \boldsymbol{q}_{n,i}^{(s)}, \boldsymbol{k}_+^{(s)} \rangle) + \sum_{l'=2}^{M} \exp(\langle \boldsymbol{q}_{n,i}^{(s)}, \boldsymbol{k}_{n,l'}^{(s)} \rangle)}
$$

$+ \{lower\ order\ term\}$

$$
\leq \frac{\eta \sigma_p^2 d}{9NM} \ell_n'^{(s)} V_+^{(s)} \cdot \frac{\exp(\langle \boldsymbol{q}_{n,i}^{(s)}, \boldsymbol{k}_{n,j}^{(s)} \rangle)}{\exp(\langle \boldsymbol{q}_{n,i}^{(s)}, \boldsymbol{k}_+^{(s)} \rangle) + \sum_{l'=2}^{M} \exp(\langle \boldsymbol{q}_{n,i}^{(s)}, \boldsymbol{k}_{n,l'}^{(s)} \rangle)}
$$

$$\cdot \frac{\exp(\langle \boldsymbol{q}_{n,i}^{(s)}, \boldsymbol{k}_{+}^{(s)}\rangle)}{\exp(\langle \boldsymbol{q}_{n,i}^{(s)}, \boldsymbol{k}_{+}^{(s)}\rangle) + \sum\limits_{l'=2}^{M} \exp(\langle \boldsymbol{q}_{n,i}^{(s)}, \boldsymbol{k}_{n,l'}^{(s)}\rangle)},$$

where the $\{lower\ order\ term\}$ is by the property that $\|\boldsymbol{\xi}_{n,i}\|_2^2$ is much larger than $\langle \boldsymbol{\xi}_{n',i'}, \boldsymbol{\xi}_{n,i}\rangle$ in Lemma C.4 and the condition $d = \widetilde{\Omega}\left(\epsilon^{-2}N^2 d_h\right)$, the third inequality is by $V_+^{(s)} \geq 3M \cdot |V_{n,i}^{(s)}|$ and $softmax(\langle \boldsymbol{q}_{n,i}^{(s)}, \boldsymbol{k}_{+}^{(s)}\rangle) \geq \frac{1}{M} - o(1)$, the last inequality is by $\|\boldsymbol{\xi}_{n,i}\|_2^2 \geq \frac{\sigma_p^2 d}{2}$ and we absorb the $\{lower\ order\ term\}$. Similarly, we have

$$\alpha_{n,i,n,j}^{(s)} \leq$$

$$-\frac{\eta \sigma_p^2 d}{9NM} \ell_n'^{(s)} V_-^{(s)} \cdot \frac{\exp(\langle \boldsymbol{q}_{n,i}^{(s)}, \boldsymbol{k}_{n,j}^{(s)}\rangle)}{\exp(\langle \boldsymbol{q}_{n,i}^{(s)}, \boldsymbol{k}_{-}^{(s)}\rangle) + \sum\limits_{l'=2}^{M} \exp(\langle \boldsymbol{q}_{n,i}^{(s)}, \boldsymbol{k}_{n,l'}^{(s)}\rangle)}$$

$$\cdot \frac{\exp(\langle \boldsymbol{q}_{n,i}^{(s)}, \boldsymbol{k}_{-}^{(s)}\rangle)}{\exp(\langle \boldsymbol{q}_{n,i}^{(s)}, \boldsymbol{k}_{-}^{(s)}\rangle) + \sum\limits_{l'=2}^{M} \exp(\langle \boldsymbol{q}_{n,i}^{(s)}, \boldsymbol{k}_{n,l'}^{(s)}\rangle)}.$$

$$\beta_{n,j,n,i}^{(s)} = \frac{\eta}{NM} - \ell_n'^{(s)}$$

$$\left(-V_+^{(s)} \sum\limits_{k=2}^{M} \left(\langle \boldsymbol{\xi}_{n,j}, \boldsymbol{\xi}_{n,k}\rangle \frac{\exp(\langle \boldsymbol{q}_{n,i}^{(s)}, \boldsymbol{k}_{+}^{(s)}\rangle)}{\exp(\langle \boldsymbol{q}_{n,i}^{(s)}, \boldsymbol{k}_{+}^{(s)}\rangle) + \sum\limits_{l'=2}^{M} \exp(\langle \boldsymbol{q}_{n,i}^{(s)}, \boldsymbol{k}_{n,l'}^{(s)}\rangle)}\right.\right.$$

$$\cdot \frac{\exp(\langle \boldsymbol{q}_{n,i}^{(s)}, \boldsymbol{k}_{n,k}^{(s)}\rangle)}{\exp(\langle \boldsymbol{q}_{n,i}^{(s)}, \boldsymbol{k}_{+}^{(s)}\rangle) + \sum\limits_{l'=2}^{M} \exp(\langle \boldsymbol{q}_{n,i}^{(s)}, \boldsymbol{k}_{n,l'}^{(s)}\rangle)}\right)$$

$$+ \sum\limits_{k=2}^{M} V_{n,k}^{(s)} \left(\langle \boldsymbol{\xi}_{n,j}, \boldsymbol{\xi}_{n,k}\rangle \frac{\exp(\langle \boldsymbol{q}_{n,i}^{(s)}, \boldsymbol{k}_{n,k}^{(s)}\rangle)}{\exp(\langle \boldsymbol{q}_{n,i}^{(s)}, \boldsymbol{k}_{+}^{(s)}\rangle) + \sum\limits_{l'=2}^{M} \exp(\langle \boldsymbol{q}_{n,i}^{(s)}, \boldsymbol{k}_{n,l'}^{(s)}\rangle)}\right.$$

$$- \sum\limits_{l=2}^{M} \left(\langle \boldsymbol{\xi}_{n,j}, \boldsymbol{\xi}_{n,l}\rangle \frac{\exp(\langle \boldsymbol{q}_{n,i}^{(s)}, \boldsymbol{k}_{n,k}^{(s)}\rangle)}{\exp(\langle \boldsymbol{q}_{n,i}^{(s)}, \boldsymbol{k}_{+}^{(s)}\rangle) + \sum\limits_{l'=2}^{M} \exp(\langle \boldsymbol{q}_{n,i}^{(s)}, \boldsymbol{k}_{n,l'}^{(s)}\rangle)}\right.$$

$$\left.\left.\left.\cdot \frac{\exp(\langle \boldsymbol{q}_{n,i}^{(s)}, \boldsymbol{k}_{n,l}^{(s)}\rangle)}{\exp(\langle \boldsymbol{q}_{n,i}^{(s)}, \boldsymbol{k}_{+}^{(s)}\rangle) + \sum\limits_{l'=2}^{M} \exp(\langle \boldsymbol{q}_{n,i}^{(s)}, \boldsymbol{k}_{n,l'}^{(s)}\rangle)}\right)\right)\right)$$

$$= \frac{\eta \|\boldsymbol{\xi}_{n,j}\|_2^2}{NM} - \ell_n'^{(s)} \cdot \frac{\exp(\langle \boldsymbol{q}_{n,i}^{(s)}, \boldsymbol{k}_{n,j}^{(s)}\rangle)}{\exp(\langle \boldsymbol{q}_{n,i}^{(s)}, \boldsymbol{k}_{+}^{(s)}\rangle) + \sum\limits_{l'=2}^{M} \exp(\langle \boldsymbol{q}_{n,i}^{(s)}, \boldsymbol{k}_{n,l'}^{(s)}\rangle)}$$

$$\left(-V_+^{(s)} \left(\frac{\exp(\langle \boldsymbol{q}_{n,i}^{(s)}, \boldsymbol{k}_{+}^{(s)}\rangle)}{\exp(\langle \boldsymbol{q}_{n,i}^{(s)}, \boldsymbol{k}_{+}^{(s)}\rangle) + \sum\limits_{l'=2}^{M} \exp(\langle \boldsymbol{q}_{n,i}^{(s)}, \boldsymbol{k}_{n,l'}^{(s)}\rangle)}\right)\right.$$

$$\left.+ V_{n,j}^{(s)} \left(1 - \frac{\exp(\langle \boldsymbol{q}_{n,i}^{(s)}, \boldsymbol{k}_{n,j}^{(s)}\rangle)}{\exp(\langle \boldsymbol{q}_{n,i}^{(s)}, \boldsymbol{k}_{+}^{(s)}\rangle) + \sum\limits_{l'=2}^{M} \exp(\langle \boldsymbol{q}_{n,i}^{(s)}, \boldsymbol{k}_{n,l'}^{(s)}\rangle)}\right)\right)$$

$$+ \{lower\ order\ term\}$$

$$\leq \frac{\eta\|\boldsymbol{\xi}_{n,j}\|_2^2}{NM} - \ell_n'^{(s)} \cdot \frac{\exp(\langle \boldsymbol{q}_{n,i}^{(s)}, \boldsymbol{k}_{n,j}^{(s)}\rangle)}{\exp(\langle \boldsymbol{q}_{n,i}^{(s)}, \boldsymbol{k}_{+}^{(s)}\rangle) + \sum\limits_{l'=2}^{M} \exp(\langle \boldsymbol{q}_{n,i}^{(s)}, \boldsymbol{k}_{n,l'}^{(s)}\rangle)}$$

$$\left(-V_{+}^{(s)} \frac{\exp(\langle \boldsymbol{q}_{n,i}^{(s)}, \boldsymbol{k}_{+}^{(s)}\rangle)}{\exp(\langle \boldsymbol{q}_{n,i}^{(s)}, \boldsymbol{k}_{+}^{(s)}\rangle) + \sum\limits_{l'=2}^{M} \exp(\langle \boldsymbol{q}_{n,i}^{(s)}, \boldsymbol{k}_{n,l'}^{(s)}\rangle)} + \max_j |V_{n,j}^{(s)}|\right)$$

$$+ \{lower\ order\ term\}$$

$$\leq \frac{\eta\|\boldsymbol{\xi}_{n,j}\|_2^2}{NM} - \ell_n'^{(s)} \cdot \frac{\exp(\langle \boldsymbol{q}_{n,i}^{(s)}, \boldsymbol{k}_{n,j}^{(s)}\rangle)}{\exp(\langle \boldsymbol{q}_{n,i}^{(s)}, \boldsymbol{k}_{+}^{(s)}\rangle) + \sum\limits_{l'=2}^{M} \exp(\langle \boldsymbol{q}_{n,i}^{(s)}, \boldsymbol{k}_{n,l'}^{(s)}\rangle)}$$

$$\left(-V_{+}^{(s)} \frac{\exp(\langle \boldsymbol{q}_{n,i}^{(s)}, \boldsymbol{k}_{+}^{(s)}\rangle)}{\exp(\langle \boldsymbol{q}_{n,i}^{(s)}, \boldsymbol{k}_{+}^{(s)}\rangle) + \sum\limits_{l'=2}^{M} \exp(\langle \boldsymbol{q}_{n,i}^{(s)}, \boldsymbol{k}_{n,l'}^{(s)}\rangle)}\right.$$

$$\left. + \frac{1}{2}V_{+}^{(s)} \frac{\exp(\langle \boldsymbol{q}_{n,i}^{(s)}, \boldsymbol{k}_{+}^{(s)}\rangle)}{\exp(\langle \boldsymbol{q}_{n,i}^{(s)}, \boldsymbol{k}_{+}^{(s)}\rangle) + \sum\limits_{l'=2}^{M} \exp(\langle \boldsymbol{q}_{n,i}^{(s)}, \boldsymbol{k}_{n,l'}^{(s)}\rangle)}\right)$$

$$+ \{lower\ order\ term\}$$

$$\leq \frac{\eta\|\boldsymbol{\xi}_{n,j}\|_2^2}{2NM} \ell_n'^{(s)} V_{+}^{(s)} \cdot \frac{\exp(\langle \boldsymbol{q}_{n,i}^{(s)}, \boldsymbol{k}_{n,j}^{(s)}\rangle)}{\exp(\langle \boldsymbol{q}_{n,i}^{(s)}, \boldsymbol{k}_{+}^{(s)}\rangle) + \sum\limits_{l'=2}^{M} \exp(\langle \boldsymbol{q}_{n,i}^{(s)}, \boldsymbol{k}_{n,l'}^{(s)}\rangle)}$$

$$\cdot \frac{\exp(\langle \boldsymbol{q}_{n,i}^{(s)}, \boldsymbol{k}_{+}^{(s)}\rangle)}{\exp(\langle \boldsymbol{q}_{n,i}^{(s)}, \boldsymbol{k}_{+}^{(s)}\rangle) + \sum\limits_{l'=2}^{M} \exp(\langle \boldsymbol{q}_{n,i}^{(s)}, \boldsymbol{k}_{n,l'}^{(s)}\rangle)}$$

$$+ \{lower\ order\ term\}$$

$$\leq \frac{\eta\sigma_p^2 d}{5NM} \ell_n'^{(s)} V_{+}^{(s)} \cdot \frac{\exp(\langle \boldsymbol{q}_{n,i}^{(s)}, \boldsymbol{k}_{n,j}^{(s)}\rangle)}{\exp(\langle \boldsymbol{q}_{n,i}^{(s)}, \boldsymbol{k}_{+}^{(s)}\rangle) + \sum\limits_{l'=2}^{M} \exp(\langle \boldsymbol{q}_{n,i}^{(s)}, \boldsymbol{k}_{n,l'}^{(s)}\rangle)}$$

$$\cdot \frac{\exp(\langle \boldsymbol{q}_{n,i}^{(s)}, \boldsymbol{k}_{+}^{(s)}\rangle)}{\exp(\langle \boldsymbol{q}_{n,i}^{(s)}, \boldsymbol{k}_{+}^{(s)}\rangle) + \sum\limits_{l'=2}^{M} \exp(\langle \boldsymbol{q}_{n,i}^{(s)}, \boldsymbol{k}_{n,l'}^{(s)}\rangle)},$$

where the $\{lower\ order\ term\}$ is by the property that $\|\boldsymbol{\xi}_{n,i}\|_2^2$ is much larger than $\langle \boldsymbol{\xi}_{n',i'}, \boldsymbol{\xi}_{n,i}\rangle$ in Lemma C.4 and the condition $d = \widetilde{\Omega}\left(\epsilon^{-2}N^2 d_h\right)$, the second inequality is by $V_{+}^{(s)} \geq 3M \cdot |V_{n,i}^{(s)}|$, the last inequality is by $\|\boldsymbol{\xi}_{n,j}\|_2^2 \geq \frac{\sigma_p^2 d}{2}$ and we absorb the $\{lower\ order\ term\}$. Similarly, we have

$$\beta_{n,j,n,i}^{(s)} \leq$$

$$- \frac{\eta\sigma_p^2 d}{5NM} \ell_n'^{(s)} V_{-}^{(s)} \cdot \frac{\exp(\langle \boldsymbol{q}_{n,i}^{(s)}, \boldsymbol{k}_{n,j}^{(s)}\rangle)}{\exp(\langle \boldsymbol{q}_{n,i}^{(s)}, \boldsymbol{k}_{-}^{(s)}\rangle) + \sum\limits_{l'=2}^{M} \exp(\langle \boldsymbol{q}_{n,i}^{(s)}, \boldsymbol{k}_{n,l'}^{(s)}\rangle)}$$

$$\cdot \frac{\exp(\langle \boldsymbol{q}_{n,i}^{(s)}, \boldsymbol{k}_{-}^{(s)}\rangle)}{\exp(\langle \boldsymbol{q}_{n,i}^{(s)}, \boldsymbol{k}_{-}^{(s)}\rangle) + \sum\limits_{l'=2}^{M} \exp(\langle \boldsymbol{q}_{n,i}^{(s)}, \boldsymbol{k}_{n,l'}^{(s)}\rangle)}.$$

Summarizing the above equations, we have the signs of $\alpha$ and $\beta$ as follows:

$$\alpha_{+,+}^{(s)}, \alpha_{-,-}^{(s)}, \beta_{+,+}^{(s)}, \beta_{-,-}^{(s)}, \alpha_{n,i,+}^{(s)}, \alpha_{n,i,-}^{(s)}, \beta_{n,+,i}^{(s)}, \beta_{n,-,i}^{(s)} \geq 0,$$

$$\alpha_{n,+,i}^{(s)}, \alpha_{n,-,i}^{(s)}, \alpha_{n,i,n,j}^{(s)}, \beta_{n,i,+}^{(s)}, \beta_{n,i,-}^{(s)}, \beta_{n,j,n,i}^{(s)} \leq 0.$$

## F.5 Lower Bounds of $\langle q, k \rangle$

In order to give the lower bounds for $\langle q, k \rangle$, we need to rewrite the bounds of $\alpha$ and $\beta$ in a more concise form. We first expand the equations in F.4 under the assumption that $\mathcal{B}(s)$ holds for $s \in [T_1, t]$.

$$\alpha_{+,+}^{(s)}$$

$$\geq \frac{\eta}{2NM} \sum_{n \in S_+} -\ell_n'^{(s)} \|\boldsymbol{\mu}\|_2^2 V_+^{(s)} \cdot \frac{\exp(\langle \boldsymbol{q}_+^{(s)}, \boldsymbol{k}_+^{(s)} \rangle)}{\exp(\langle \boldsymbol{q}_+^{(s)}, \boldsymbol{k}_+^{(s)} \rangle) + \sum\limits_{j'=2}^{M} \exp(\langle \boldsymbol{q}_+^{(s)}, \boldsymbol{k}_{n,j'}^{(s)} \rangle)}$$

$$\cdot \left(1 - \frac{\exp(\langle \boldsymbol{q}_+^{(s)}, \boldsymbol{k}_+^{(s)} \rangle)}{\exp(\langle \boldsymbol{q}_+^{(s)}, \boldsymbol{k}_+^{(s)} \rangle) + \sum\limits_{j'=2}^{M} \exp(\langle \boldsymbol{q}_+^{(s)}, \boldsymbol{k}_{n,j'}^{(s)} \rangle)}\right)$$

$$= \frac{\eta}{2NM} \sum_{n \in S_+} -\ell_n'^{(s)} \|\boldsymbol{\mu}\|_2^2 V_+^{(s)} \cdot \frac{\exp(\langle \boldsymbol{q}_+^{(s)}, \boldsymbol{k}_+^{(s)} \rangle)}{\exp(\langle \boldsymbol{q}_+^{(s)}, \boldsymbol{k}_+^{(s)} \rangle) + \sum\limits_{j'=2}^{M} \exp(\langle \boldsymbol{q}_+^{(s)}, \boldsymbol{k}_{n,j'}^{(s)} \rangle)}$$

$$\cdot \frac{\sum\limits_{j'=2}^{M} \exp(\langle \boldsymbol{q}_+^{(s)}, \boldsymbol{k}_{n,j'}^{(s)} \rangle)}{\exp(\langle \boldsymbol{q}_+^{(s)}, \boldsymbol{k}_+^{(s)} \rangle) + \sum\limits_{j'=2}^{M} \exp(\langle \boldsymbol{q}_+^{(s)}, \boldsymbol{k}_{n,j'}^{(s)} \rangle)}$$

$$\geq \frac{\eta}{2NM} - \ell_n'^{(s)} \|\boldsymbol{\mu}\|_2^2 V_+^{(s)} \cdot \frac{\exp(\langle \boldsymbol{q}_+^{(s)}, \boldsymbol{k}_+^{(s)} \rangle)}{\exp(\langle \boldsymbol{q}_+^{(s)}, \boldsymbol{k}_+^{(s)} \rangle) + \sum\limits_{j'=2}^{M} \exp(\langle \boldsymbol{q}_+^{(s)}, \boldsymbol{k}_{n,j'}^{(s)} \rangle)} \qquad (223)$$

$$\cdot \frac{\exp(\langle \boldsymbol{q}_+^{(s)}, \boldsymbol{k}_{n,j}^{(s)} \rangle)}{\exp(\langle \boldsymbol{q}_+^{(s)}, \boldsymbol{k}_+^{(s)} \rangle) + \sum\limits_{j'=2}^{M} \exp(\langle \boldsymbol{q}_+^{(s)}, \boldsymbol{k}_{n,j'}^{(s)} \rangle)}$$

$$\geq \frac{\eta}{2NM} - \ell_n'^{(s)} \|\boldsymbol{\mu}\|_2^2 V_+^{(s)} \cdot \left(\frac{1}{M} - o(1)\right)$$

$$\cdot \frac{\exp(\langle \boldsymbol{q}_+^{(s)}, \boldsymbol{k}_{n,j}^{(s)} \rangle)}{\exp(\langle \boldsymbol{q}_+^{(s)}, \boldsymbol{k}_+^{(s)} \rangle) + \sum\limits_{j'=2}^{M} \exp(\langle \boldsymbol{q}_+^{(s)}, \boldsymbol{k}_{n,j'}^{(s)} \rangle)}$$

$$\geq \frac{\eta}{2NM} - \ell_n'^{(s)} \|\boldsymbol{\mu}\|_2^2 V_+^{(s)} \cdot \left(\frac{1}{M} - o(1)\right) \cdot \frac{\exp(\langle \boldsymbol{q}_+^{(s)}, \boldsymbol{k}_{n,j}^{(s)} \rangle)}{C \exp(\langle \boldsymbol{q}_+^{(s)}, \boldsymbol{k}_+^{(s)} \rangle)}$$

$$\geq \frac{\eta^2 C_5 \|\boldsymbol{\mu}\|_2^4 \|\boldsymbol{w}_O\|_2^2 (s - T_1)}{N} \cdot \frac{1}{\exp(\Lambda_{n,+,j}^{(s)})},$$

where the $\langle \boldsymbol{q}_+^{(s)}, \boldsymbol{k}_{n,j}^{(s)} \rangle$ in the second inequality is a particular choice (we will characterize the dynamic of $\langle \boldsymbol{q}_+^{(s)}, \boldsymbol{k}_+^{(s)} \rangle - \langle \boldsymbol{q}_+^{(s)}, \boldsymbol{k}_{n,j}^{(s)} \rangle$), the third inequality is by $softmax(\langle \boldsymbol{q}_+^{(s)}, \boldsymbol{k}_+^{(s)} \rangle) \geq \left(\frac{1}{M} - o(1)\right)$. In the fourth inequality, by $\langle \boldsymbol{q}^{(T_1)}, \boldsymbol{k}^{(T_1)} \rangle = o(1)$ and the monotonicity of $\langle \boldsymbol{q}^{(s)}, \boldsymbol{k}^{(s)} \rangle$ ($\langle \boldsymbol{q}_+^{(s)}, \boldsymbol{k}_+^{(s)} \rangle$ is increasing and $\langle \boldsymbol{q}_+^{(s)}, \boldsymbol{k}_{n,j}^{(s)} \rangle$ is decreasing), there exist a constant C such that $C \exp(\langle \boldsymbol{q}_+^{(s)}, \boldsymbol{k}_+^{(s)} \rangle) \geq \exp(\langle \boldsymbol{q}_+^{(s)}, \boldsymbol{k}_+^{(s)} \rangle) + \sum\limits_{j'=2}^{M} \exp(\langle \boldsymbol{q}_+^{(s)}, \boldsymbol{k}_{n,j'}^{(s)} \rangle)$. In the last inequality, we plugging the lower bounds of

$V_+^{(s)}$ and $-\ell_n'^{(s)}$ and then absorb all the constant factors. Similarly, we have

$$\beta_{+,+}^{(s)} \geq \frac{\eta^2 C_5 \|\boldsymbol{\mu}\|_2^4 \|\boldsymbol{w}_O\|_2^2 (s - T_1)}{N} \cdot \frac{1}{\exp(\Lambda_{n,+,j}^{(s)})}, \tag{224}$$

$$\alpha_{-,-}^{(s)} \geq \frac{\eta^2 C_5 \|\boldsymbol{\mu}\|_2^4 \|\boldsymbol{w}_O\|_2^2 (s - T_1)}{N} \cdot \frac{1}{\exp(\Lambda_{n,-,j}^{(s)})}, \tag{225}$$

$$\beta_{-,-}^{(s)} \geq \frac{\eta^2 C_5 \|\boldsymbol{\mu}\|_2^4 \|\boldsymbol{w}_O\|_2^2 (s - T_1)}{N} \cdot \frac{1}{\exp(\Lambda_{n,-,j}^{(s)})}. \tag{226}$$

$$
\begin{aligned}
&\alpha_{n,+,j}^{(s)} \\
&\leq \frac{\eta}{4NM} \ell_n'^{(s)} \|\boldsymbol{\mu}\|_2^2 V_+^{(s)} \cdot \frac{\exp(\langle \boldsymbol{q}_+^{(s)}, \boldsymbol{k}_{n,j}^{(s)} \rangle)}{\exp(\langle \boldsymbol{q}_+^{(s)}, \boldsymbol{k}_+^{(s)} \rangle) + \sum\limits_{j'=2}^{M} \exp(\langle \boldsymbol{q}_+^{(s)}, \boldsymbol{k}_{n,j'}^{(s)} \rangle)} \\
&\quad \cdot \frac{\exp(\langle \boldsymbol{q}_+^{(s)}, \boldsymbol{k}_+^{(s)} \rangle)}{\exp(\langle \boldsymbol{q}_+^{(s)}, \boldsymbol{k}_+^{(s)} \rangle) + \sum\limits_{j'=2}^{M} \exp(\langle \boldsymbol{q}_+^{(s)}, \boldsymbol{k}_{n,j'}^{(s)} \rangle)} \\
&\leq \frac{\eta}{4NM} \ell_n'^{(s)} \|\boldsymbol{\mu}\|_2^2 V_+^{(s)} \cdot \left(\frac{1}{M} - o(1)\right) \\
&\quad \cdot \frac{\exp(\langle \boldsymbol{q}_+^{(s)}, \boldsymbol{k}_{n,j}^{(s)} \rangle)}{\exp(\langle \boldsymbol{q}_+^{(s)}, \boldsymbol{k}_+^{(s)} \rangle) + \sum\limits_{j'=2}^{M} \exp(\langle \boldsymbol{q}_+^{(s)}, \boldsymbol{k}_{n,j'}^{(s)} \rangle)} \\
&\leq \frac{\eta}{4NM} \ell_n'^{(s)} \|\boldsymbol{\mu}\|_2^2 V_+^{(s)} \cdot \left(\frac{1}{M} - o(1)\right) \cdot \frac{\exp(\langle \boldsymbol{q}_+^{(s)}, \boldsymbol{k}_{n,j}^{(s)} \rangle)}{C \exp(\langle \boldsymbol{q}_+^{(s)}, \boldsymbol{k}_+^{(s)} \rangle)} \\
&\leq -\frac{\eta^2 C_5 \|\boldsymbol{\mu}\|_2^4 \|\boldsymbol{w}_O\|_2^2 (s - T_1)}{N} \cdot \frac{1}{\exp(\Lambda_{n,+,j}^{(s)})},
\end{aligned}
\tag{227}
$$

where the inequalities is similar to (223). Similarly, we have

$$\alpha_{n,-,j}^{(s)} \leq -\frac{\eta^2 C_5 \|\boldsymbol{\mu}\|_2^4 \|\boldsymbol{w}_O\|_2^2 (s - T_1)}{N} \cdot \frac{1}{\exp(\Lambda_{n,-,j}^{(s)})}.$$

$$\beta_{n,+,i}^{(s)}$$

$$\geq -\frac{\eta\|\boldsymbol{\mu}\|_2^2}{2NM}\ell_n'^{(s)}V_+^{(s)}\frac{\exp(\langle\boldsymbol{q}_{n,i}^{(s)},\boldsymbol{k}_+^{(s)}\rangle)}{\exp(\langle\boldsymbol{q}_{n,i}^{(s)},\boldsymbol{k}_+^{(s)}\rangle)+\sum\limits_{j'=2}^{M}\exp(\langle\boldsymbol{q}_{n,i}^{(s)},\boldsymbol{k}_{n,j'}^{(s)}\rangle)}$$

$$\cdot\Big(1-\frac{\exp(\langle\boldsymbol{q}_{n,i}^{(s)},\boldsymbol{k}_+^{(s)}\rangle)}{\exp(\langle\boldsymbol{q}_{n,i}^{(s)},\boldsymbol{k}_+^{(s)}\rangle)+\sum\limits_{j'=2}^{M}\exp(\langle\boldsymbol{q}_{n,i}^{(s)},\boldsymbol{k}_{n,j'}^{(s)}\rangle)}\Big)$$

$$\geq -\frac{\eta\|\boldsymbol{\mu}\|_2^2}{2NM}\ell_n'^{(s)}V_+^{(s)}\cdot\Big(\frac{1}{M}-o(1)\Big)$$

$$\cdot\frac{\sum\limits_{j'=2}^{M}\exp(\langle\boldsymbol{q}_{n,i}^{(s)},\boldsymbol{k}_{n,j'}^{(s)}\rangle)}{\exp(\langle\boldsymbol{q}_{n,i}^{(s)},\boldsymbol{k}_+^{(s)}\rangle)+\sum\limits_{j'=2}^{M}\exp(\langle\boldsymbol{q}_{n,i}^{(s)},\boldsymbol{k}_{n,j'}^{(s)}\rangle)} \qquad (228)$$

$$\geq -\frac{\eta\|\boldsymbol{\mu}\|_2^2}{2NM}\ell_n'^{(s)}V_+^{(s)}\cdot\Big(\frac{1}{M}-o(1)\Big)$$

$$\cdot\frac{\exp(\langle\boldsymbol{q}_{n,i}^{(s)},\boldsymbol{k}_{n,j}^{(s)}\rangle)}{\exp(\langle\boldsymbol{q}_{n,i}^{(s)},\boldsymbol{k}_+^{(s)}\rangle)+\sum\limits_{j'=2}^{M}\exp(\langle\boldsymbol{q}_{n,i}^{(s)},\boldsymbol{k}_{n,j'}^{(s)}\rangle)}$$

$$\geq -\frac{\eta\|\boldsymbol{\mu}\|_2^2}{2NM}\ell_n'^{(s)}V_+^{(s)}\cdot\Big(\frac{1}{M}-o(1)\Big)\cdot\frac{\exp(\langle\boldsymbol{q}_{n,i}^{(s)},\boldsymbol{k}_{n,j}^{(s)}\rangle)}{C\exp(\langle\boldsymbol{q}_{n,i}^{(s)},\boldsymbol{k}_+^{(s)}\rangle)}$$

$$\geq \frac{\eta^2 C_5\|\boldsymbol{\mu}\|_2^4\|\boldsymbol{w}_O\|_2^2(s-T_1)}{N}\cdot\frac{1}{\exp(\Lambda_{n,i,+,j}^{(s)})},$$

where the inequalities is similar to (223), and $\langle\boldsymbol{q}_{n,i}^{(s)},\boldsymbol{k}_{n,j}^{(s)}\rangle$ is a particular choice (we will characterize the dynamic of $\langle\boldsymbol{q}_{n,i}^{(s)},\boldsymbol{k}_+^{(s)}\rangle - \langle\boldsymbol{q}_{n,i}^{(s)},\boldsymbol{k}_{n,j}^{(s)}\rangle$). Similarly, we have

$$\beta_{n,-,i}^{(s)} \geq \frac{\eta^2 C_5\|\boldsymbol{\mu}\|_2^4\|\boldsymbol{w}_O\|_2^2(s-T_1)}{N}\cdot\frac{1}{\exp(\Lambda_{n,i,-,j}^{(s)})}, \qquad (229)$$

$$\alpha_{n,i,+}^{(s)} \geq \frac{\eta^2 C_5\sigma_p^2 d\|\boldsymbol{\mu}\|_2^2\|\boldsymbol{w}_O\|_2^2(s-T_1)}{N}\cdot\frac{1}{\exp(\Lambda_{n,i,+,j}^{(s)})}, \qquad (230)$$

$$\alpha_{n,i,-}^{(s)} \geq \frac{\eta^2 C_5\sigma_p^2 d\|\boldsymbol{\mu}\|_2^2\|\boldsymbol{w}_O\|_2^2(s-T_1)}{N}\cdot\frac{1}{\exp(\Lambda_{n,i,-,j}^{(s)})}, \qquad (231)$$

$$\beta_{n,j,+}^{(s)} \leq -\frac{\eta^2 C_5\sigma_p^2 d\|\boldsymbol{\mu}\|_2^2\|\boldsymbol{w}_O\|_2^2(s-T_1)}{N}\cdot\frac{1}{\exp(\Lambda_{n,+,j}^{(s)})}, \qquad (232)$$

$$\beta_{n,j,-}^{(s)} \leq -\frac{\eta^2 C_5\sigma_p^2 d\|\boldsymbol{\mu}\|_2^2\|\boldsymbol{w}_O\|_2^2(s-T_1)}{N}\cdot\frac{1}{\exp(\Lambda_{n,-,j}^{(s)})}, \qquad (233)$$

$$\alpha_{n,i,n,j}^{(s)} \leq -\frac{\eta^2 C_5\sigma_p^2 d\|\boldsymbol{\mu}\|_2^2\|\boldsymbol{w}_O\|_2^2(s-T_1)}{N}\cdot\frac{1}{\exp(\Lambda_{n,i,\pm,j}^{(s)})}, \qquad (234)$$

$$\beta_{n,j,n,i}^{(s)} \leq -\frac{\eta^2 C_5\sigma_p^2 d\|\boldsymbol{\mu}\|_2^2\|\boldsymbol{w}_O\|_2^2(s-T_1)}{N}\cdot\frac{1}{\exp(\Lambda_{n,i,\pm,j}^{(s)})}. \qquad (235)$$

With the concise lower bounds for $\alpha$ and $\beta$ above and proposition $\mathcal{C}(s)$, we will give the lower bounds for the dynamics of $\langle q, k \rangle$.

$$
\begin{aligned}
&\langle q_+^{(s+1)}, k_+^{(s+1)} \rangle - \langle q_+^{(s)}, k_+^{(s)} \rangle \\
&= \alpha_{+,+}^{(s)} \| k_+^{(s)} \|_2^2 + \sum_{n \in S_+} \sum_{i=2}^{M} \alpha_{n,+,i}^{(s)} \langle k_+^{(s)}, k_{n,i}^{(s)} \rangle \\
&\quad + \beta_{+,+}^{(s)} \| q_+^{(s)} \|_2^2 + \sum_{n \in S_+} \sum_{i=2}^{M} \beta_{n,+,i}^{(s)} \langle q_+^{(s)}, q_{n,i}^{(s)} \rangle \\
&\quad + \left( \alpha_{+,+}^{(s)} k_+^{(s)} + \sum_{n \in S_+} \sum_{i=2}^{M} \alpha_{n,+,i}^{(s)} k_{n,i}^{(s)} \right) \\
&\quad \cdot \left( \beta_{+,+}^{(s)} q_+^{(s)\top} + \sum_{n \in S_+} \sum_{i=2}^{M} \beta_{n,+,i}^{(s)} q_{n,i}^{(s)\top} \right) \\
&= \alpha_{+,+}^{(s)} \| k_+^{(s)} \|_2^2 + \beta_{+,+}^{(s)} \| q_+^{(s)} \|_2^2 + \{lower\ order\ term\} \\
&\geq \frac{2\eta^2 C_5 \| \mu \|_2^4 \| w_O \|_2^2 (s - T_1)}{N} \cdot \frac{1}{\exp(\Lambda_{n,+,j}^{(s)})} \cdot \Theta(\| \mu \|_2^2 \sigma_h^2 d_h) \\
&\quad + \{lower\ order\ term\} \\
&\geq \frac{\eta^2 C_6 \| \mu \|_2^6 \| w_O \|_2^2 \sigma_h^2 d_h (s - T_1)}{N} \cdot \frac{1}{\exp(\Lambda_{n,+,j}^{(s)})},
\end{aligned}
\tag{236}
$$

where the first inequality is by (223), (224), the bound of $\| q_+^{(s)} \|_2^2$, $\| k_+^{(s)} \|_2^2$ in stage II, the second inequality is by absorbing the $\{lower\ order\ term\}$ and the constant factors. Similarly, we have

$$
\begin{aligned}
&\langle q_-^{(s+1)}, k_-^{(s+1)} \rangle - \langle q_-^{(s)}, k_-^{(s)} \rangle \\
&\geq \frac{\eta^2 C_6 \| \mu \|_2^6 \| w_O \|_2^2 \sigma_h^2 d_h (s - T_1)}{N} \cdot \frac{1}{\exp(\Lambda_{n,-,j}^{(s)})}.
\end{aligned}
\tag{237}
$$

$$\langle \boldsymbol{q}_+^{(s+1)}, \boldsymbol{k}_{n,j}^{(s+1)} \rangle - \langle \boldsymbol{q}_+^{(s)}, \boldsymbol{k}_{n,j}^{(s)} \rangle$$

$$= \alpha_{+,+}^{(s)} \langle \boldsymbol{k}_+^{(s)}, \boldsymbol{k}_{n,j}^{(s)} \rangle + \sum_{n' \in S_+} \sum_{l=2}^{M} \alpha_{n',+,l}^{(s)} \langle \boldsymbol{k}_{n,j}^{(s)}, \boldsymbol{k}_{n',l}^{(s)} \rangle$$

$$+ \beta_{n,j,+}^{(s)} \|\boldsymbol{q}_+^{(s)}\|_2^2 + \beta_{n,j,-}^{(s)} \langle \boldsymbol{q}_+^{(s)}, \boldsymbol{q}_-^{(s)} \rangle + \sum_{n'=1}^{N} \sum_{l=2}^{M} \beta_{n,j,n',l}^{(s)} \langle \boldsymbol{q}_+^{(s)}, \boldsymbol{q}_{n',l}^{(s)} \rangle$$

$$+ \left( \alpha_{+,+}^{(s)} \boldsymbol{k}_+^{(s)} + \sum_{n' \in S_+} \sum_{l=2}^{M} \alpha_{n',+,l}^{(s)} \boldsymbol{k}_{n',l}^{(s)} \right)$$

$$\cdot \left( \beta_{n,j,+}^{(s)} \boldsymbol{q}_+^{(s)\top} + \beta_{n,j,-}^{(s)} \boldsymbol{q}_-^{(s)\top} + \sum_{n'=1}^{N} \sum_{l=2}^{M} \beta_{n,j,n',l}^{(s)} \boldsymbol{q}_{n',l}^{(s)\top} \right) \tag{238}$$

$$= \alpha_{n,+,j}^{(s)} \|\boldsymbol{k}_{n,j}^{(s)}\|_2^2 + \beta_{n,j,+}^{(s)} \|\boldsymbol{q}_+^{(s)}\|_2^2 + \{lower\ order\ term\}$$

$$\leq -\frac{\eta^2 C_5 \|\boldsymbol{\mu}\|_2^4 \|\boldsymbol{w}_O\|_2^2 (s - T_1)}{N} \cdot \frac{1}{\exp(\Lambda_{n,+,j}^{(s)})} \cdot \Theta\left( \sigma_p^2 \sigma_h^2 d d_h \right)$$

$$- \frac{\eta^2 C_5 \sigma_p^2 d \|\boldsymbol{\mu}\|_2^2 \|\boldsymbol{w}_O\|_2^2 (s - T_1)}{N} \cdot \frac{1}{\exp(\Lambda_{n,+,j}^{(s)})} \cdot \Theta\left( \|\boldsymbol{\mu}\|_2^2 \sigma_h^2 d_h \right)$$

$$+ \{lower\ order\ term\}$$

$$\leq -\frac{\eta^2 C_6 \sigma_p^2 d \|\boldsymbol{\mu}\|_2^4 \|\boldsymbol{w}_O\|_2^2 \sigma_h^2 d_h (s - T_1)}{N} \cdot \frac{1}{\exp(\Lambda_{n,+,j}^{(s)})},$$

where the first inequality is by (227), (232), the bound of $\|\boldsymbol{k}_{n,j}^{(s)}\|_2^2$, $\|\boldsymbol{q}_+^{(s)}\|_2^2$ in stage II, the second inequality is by absorbing the $\{lower\ order\ term\}$ and the constant factors. Similarly, we have

$$\langle \boldsymbol{q}_-^{(s+1)}, \boldsymbol{k}_{n,j}^{(s+1)} \rangle - \langle \boldsymbol{q}_-^{(s)}, \boldsymbol{k}_{n,j}^{(s)} \rangle$$
$$\leq -\frac{\eta^2 C_6 \sigma_p^2 d \|\boldsymbol{\mu}\|_2^4 \|\boldsymbol{w}_O\|_2^2 \sigma_h^2 d_h (s - T_1)}{N} \cdot \frac{1}{\exp(\Lambda_{n,-,j}^{(s)})}. \tag{239}$$

$$\langle \boldsymbol{q}_{n,i}^{(s+1)}, \boldsymbol{k}_{+}^{(s+1)} \rangle - \langle \boldsymbol{q}_{n,i}^{(s)}, \boldsymbol{k}_{+}^{(s)} \rangle$$

$$= \alpha_{n,i,+}^{(s)} \|\boldsymbol{k}_{+}^{(s)}\|_2^2 + \alpha_{n,i,-}^{(s)} \langle \boldsymbol{k}_{+}^{(s)}, \boldsymbol{k}_{-}^{(s)} \rangle + \sum_{n'=1}^{N} \sum_{l=2}^{M} \alpha_{n,i,n',l}^{(s)} \langle \boldsymbol{k}_{+}^{(s)}, \boldsymbol{k}_{n',l}^{(s)} \rangle$$

$$+ \beta_{+,+}^{(s)} \langle \boldsymbol{q}_{+}^{(s)}, \boldsymbol{q}_{n,i}^{(s)} \rangle + \sum_{n' \in S_+} \sum_{l=2}^{M} \beta_{n',+,l}^{(s)} \langle \boldsymbol{q}_{n,i}^{(s)}, \boldsymbol{q}_{n',l}^{(s)} \rangle$$

$$+ \left( \alpha_{n,i,+}^{(s)} \boldsymbol{k}_{+}^{(s)} + \alpha_{n,i,-}^{(s)} \boldsymbol{k}_{-}^{(s)} + \sum_{n'=1}^{N} \sum_{l=2}^{M} \alpha_{n,i,n',l}^{(s)} \boldsymbol{k}_{n',l}^{(s)} \right)$$

$$\cdot \left( \beta_{+,+}^{(s)} \boldsymbol{q}_{+}^{(s)\top} + \sum_{n' \in S_+} \sum_{l=2}^{M} \beta_{n',+,l}^{(s)} \boldsymbol{q}_{n',l}^{(s)\top} \right) \tag{240}$$

$$= \alpha_{n,i,+}^{(s)} \|\boldsymbol{k}_{+}^{(s)}\|_2^2 + \beta_{n,+,i}^{(s)} \|\boldsymbol{q}_{n,i}^{(s)}\|_2^2 + \{lower\ order\ term\}$$

$$\geq \frac{\eta^2 C_5 \sigma_p^2 d \|\boldsymbol{\mu}\|_2^2 \|\boldsymbol{w}_O\|_2^2 (s - T_1)}{N} \cdot \frac{1}{\exp(\Lambda_{n,i,+,j}^{(s)})} \cdot \Theta\left( \|\boldsymbol{\mu}\|_2^2 \sigma_h^2 d_h \right)$$

$$+ \frac{\eta^2 C_5 \|\boldsymbol{\mu}\|_2^4 \|\boldsymbol{w}_O\|_2^2 (s - T_1)}{N} \cdot \frac{1}{\exp(\Lambda_{n,i,+,j}^{(s)})} \cdot \Theta\left( \sigma_p^2 \sigma_h^2 d d_h \right)$$

$$+ \{lower\ order\ term\}$$

$$\geq \frac{\eta^2 C_6 \sigma_p^2 d \|\boldsymbol{\mu}\|_2^4 \|\boldsymbol{w}_O\|_2^2 \sigma_h^2 d_h (s - T_1)}{N} \cdot \frac{1}{\exp(\Lambda_{n,i,+,j}^{(s)})},$$

where the first inequality is by (230), (228), the bound of $\|\boldsymbol{k}_{+}^{(s)}\|_2^2$, $\|\boldsymbol{q}_{n,i}^{(s)}\|_2^2$ in stage II, the second inequality is by absorbing the $\{lower\ order\ term\}$ and the constant factors. Similarly, we have

$$\langle \boldsymbol{q}_{n,i}^{(s+1)}, \boldsymbol{k}_{-}^{(s+1)} \rangle - \langle \boldsymbol{q}_{n,i}^{(s)}, \boldsymbol{k}_{-}^{(s)} \rangle$$

$$\geq \frac{\eta^2 C_6 \sigma_p^2 d \|\boldsymbol{\mu}\|_2^4 \|\boldsymbol{w}_O\|_2^2 \sigma_h^2 d_h (s - T_1)}{N} \cdot \frac{1}{\exp(\Lambda_{n,i,-,j}^{(s)})}. \tag{241}$$

$$\langle \boldsymbol{q}_{n,i}^{(s+1)}, \boldsymbol{k}_{n,j}^{(s+1)} \rangle - \langle \boldsymbol{q}_{n,i}^{(s)}, \boldsymbol{k}_{n,j}^{(s)} \rangle$$

$$= \alpha_{n,i,+}^{(s)} \langle \boldsymbol{k}_{+}^{(s)}, \boldsymbol{k}_{n,j}^{(s)} \rangle + \alpha_{n,i,-}^{(s)} \langle \boldsymbol{k}_{-}^{(s)}, \boldsymbol{k}_{n,j}^{(s)} \rangle + \sum_{n'=1}^{N} \sum_{l=2}^{M} \alpha_{n,i,n',l}^{(s)} \langle \boldsymbol{k}_{n',l}^{(s)}, \boldsymbol{k}_{n,j}^{(s)} \rangle$$

$$+ \beta_{n,j,+}^{(s)} \langle \boldsymbol{q}_{+}^{(s)}, \boldsymbol{q}_{n,i}^{(s)} \rangle + \beta_{n,j,-}^{(s)} \langle \boldsymbol{q}_{-}^{(s)}, \boldsymbol{q}_{n,i}^{(s)} \rangle + \sum_{n'=1}^{N} \sum_{l=2}^{M} \beta_{n,j,n',l}^{(s)} \langle \boldsymbol{q}_{n',l}^{(s)}, \boldsymbol{q}_{n,i}^{(s)} \rangle$$

$$+ \left( \alpha_{n,i,+}^{(s)} \boldsymbol{k}_{+}^{(s)} + \alpha_{n,i,-}^{(s)} \boldsymbol{k}_{-}^{(s)} + \sum_{n'=1}^{N} \sum_{l=2}^{M} \alpha_{n,i,n',l}^{(s)} \boldsymbol{k}_{n',l}^{(s)} \right)$$

$$\cdot \left( \beta_{n,j,+}^{(s)} \boldsymbol{q}_{+}^{(s)\top} + \beta_{n,j,-}^{(s)} \boldsymbol{q}_{-}^{(s)\top} + \sum_{n'=1}^{N} \sum_{l=2}^{M} \beta_{n,j,n',l}^{(s)} \boldsymbol{q}_{n',l}^{(s)\top} \right) \tag{242}$$

$$= \alpha_{n,i,n,j}^{(s)} \|\boldsymbol{k}_{n,j}^{(s)}\|_2^2 + \beta_{n,j,n,i}^{(s)} \|\boldsymbol{q}_{n,i}^{(s)}\|_2^2$$

$$+ \{lower\ order\ term\}$$

$$\leq -\frac{2\eta^2 C_5 \sigma_p^2 d \|\boldsymbol{\mu}\|_2^2 \|\boldsymbol{w}_O\|_2^2 \sigma_h^2 d_h (s - T_1)}{N} \cdot \frac{1}{\exp(\Lambda_{n,i,\pm,j}^{(s)})} \cdot \Theta\left( \sigma_p^2 \sigma_h^2 d d_h \right)$$

$$+ \{lower\ order\ term\}$$

$$\leq -\frac{\eta^2 C_6 \sigma_p^4 d^2 \|\boldsymbol{\mu}\|_2^2 \|\boldsymbol{w}_O\|_2^2 \sigma_h^2 d_h (s - T_1)}{N} \cdot \frac{1}{\exp(\Lambda_{n,i,\pm,j}^{(s)})},$$

where the first inequality is by (234), (235), the bound of $\|\boldsymbol{k}_{n,j}^{(s)}\|_2^2$, $\|\boldsymbol{q}_{n,i}^{(s)}\|_2^2$ in stage II, the second inequality is by absorbing the $\{lower\ order\ term\}$ and the constant factors.

### F.6 Upper Bounds of $\langle\, \mathbf{q}, \mathbf{k}\, \rangle$

In order to give the upper bounds of $\langle \boldsymbol{q}, \boldsymbol{k} \rangle$ in stage II, we need to give the upper bounds of $\alpha$ and $\beta$ based on the equations in F.1 under the assumption that $\mathcal{D}(T_1), \ldots, \mathcal{D}(s-1)$ hold for $s \in [T_1, t]$.

$$\alpha_{+,+}^{(s)} = \frac{\eta}{NM} \sum_{n \in S_+} -\ell_n'^{(s)} \|\boldsymbol{\mu}\|_2^2$$

$$\cdot \left( V_+^{(s)} \left( \frac{\exp(\langle \boldsymbol{q}_+^{(s)}, \boldsymbol{k}_+^{(s)} \rangle)}{\exp(\langle \boldsymbol{q}_+^{(s)}, \boldsymbol{k}_+^{(s)} \rangle) + \sum_{j=2}^{M} \exp(\langle \boldsymbol{q}_+^{(s)}, \boldsymbol{k}_{n,j}^{(s)} \rangle)} \right. \right.$$

$$\left. - \left( \frac{\exp(\langle \boldsymbol{q}_+^{(s)}, \boldsymbol{k}_+^{(s)} \rangle)}{\exp(\langle \boldsymbol{q}_+^{(s)}, \boldsymbol{k}_+^{(s)} \rangle) + \sum_{j=2}^{M} \exp(\langle \boldsymbol{q}_+^{(s)}, \boldsymbol{k}_{n,j}^{(s)} \rangle)} \right)^2 \right)$$

$$- \sum_{i=2}^{M} \left( V_{n,i}^{(s)} \cdot \frac{\exp(\langle \boldsymbol{q}_+^{(s)}, \boldsymbol{k}_+^{(s)} \rangle)}{\exp(\langle \boldsymbol{q}_+^{(s)}, \boldsymbol{k}_+^{(s)} \rangle) + \sum_{j=2}^{M} \exp(\langle \boldsymbol{q}_+^{(s)}, \boldsymbol{k}_{n,j}^{(s)} \rangle)} \right.$$

$$\left. \left. \cdot \frac{\exp(\langle \boldsymbol{q}_+^{(s)}, \boldsymbol{k}_{n,i}^{(s)} \rangle)}{\exp(\langle \boldsymbol{q}_+^{(s)}, \boldsymbol{k}_+^{(s)} \rangle) + \sum_{j=2}^{M} \exp(\langle \boldsymbol{q}_+^{(s)}, \boldsymbol{k}_{n,j}^{(s)} \rangle)} \right) \right)$$

$$= \frac{\eta}{NM} \sum_{n \in S_+} -\ell_n'^{(s)} \|\boldsymbol{\mu}\|_2^2 \frac{\exp(\langle \boldsymbol{q}_+^{(s)}, \boldsymbol{k}_+^{(s)} \rangle)}{\exp(\langle \boldsymbol{q}_+^{(s)}, \boldsymbol{k}_+^{(s)} \rangle) + \sum_{j=2}^{M} \exp(\langle \boldsymbol{q}_+^{(s)}, \boldsymbol{k}_{n,j}^{(s)} \rangle)}$$

$$\cdot \left( V_+^{(s)} \cdot \frac{\sum_{j=2}^{M} \exp(\langle \boldsymbol{q}_+^{(s)}, \boldsymbol{k}_{n,j}^{(s)} \rangle)}{\exp(\langle \boldsymbol{q}_+^{(s)}, \boldsymbol{k}_+^{(s)} \rangle) + \sum_{j=2}^{M} \exp(\langle \boldsymbol{q}_+^{(s)}, \boldsymbol{k}_{n,j}^{(s)} \rangle)} \right.$$

$$\left. - \sum_{i=2}^{M} V_{n,i}^{(s)} \cdot \frac{\exp(\langle \boldsymbol{q}_+^{(s)}, \boldsymbol{k}_{n,i}^{(s)} \rangle)}{\exp(\langle \boldsymbol{q}_+^{(s)}, \boldsymbol{k}_+^{(s)} \rangle) + \sum_{j=2}^{M} \exp(\langle \boldsymbol{q}_+^{(s)}, \boldsymbol{k}_{n,j}^{(s)} \rangle)} \right)$$

$$\leq \frac{\eta}{NM} \sum_{n \in S_+} \|\boldsymbol{\mu}\|_2^2 \cdot \left( V_+^{(s)} \cdot \frac{\sum_{j=2}^{M} \exp(\langle \boldsymbol{q}_+^{(s)}, \boldsymbol{k}_{n,j}^{(s)} \rangle)}{\exp(\langle \boldsymbol{q}_+^{(s)}, \boldsymbol{k}_+^{(s)} \rangle) + \sum_{j=2}^{M} \exp(\langle \boldsymbol{q}_+^{(s)}, \boldsymbol{k}_{n,j}^{(s)} \rangle)} \right.$$

$$\left. + \max_i |V_{n,i}^{(s)}| \cdot \frac{\sum_{j=2}^{M} \exp(\langle \boldsymbol{q}_+^{(s)}, \boldsymbol{k}_{n,j}^{(s)} \rangle)}{\exp(\langle \boldsymbol{q}_+^{(s)}, \boldsymbol{k}_+^{(s)} \rangle) + \sum_{j=2}^{M} \exp(\langle \boldsymbol{q}_+^{(s)}, \boldsymbol{k}_{n,j}^{(s)} \rangle)} \right)$$

$$\leq \frac{\eta}{NM} \cdot \frac{3N}{4} \cdot \|\boldsymbol{\mu}\|_2^2 \cdot \left( V_+^{(s)} \cdot \frac{C}{\exp(\langle \boldsymbol{q}_+^{(s)}, \boldsymbol{k}_+^{(s)} \rangle)} + \max_i |V_{n,i}^{(s)}| \cdot \frac{C}{\exp(\langle \boldsymbol{q}_+^{(s)}, \boldsymbol{k}_+^{(s)} \rangle)} \right)$$

$$\leq \frac{\eta C_9 \|\boldsymbol{\mu}\|_2^2}{\exp(\langle \boldsymbol{q}_+^{(s)}, \boldsymbol{k}_+^{(s)} \rangle)},$$

where the first inequality is by $-\ell_n'^{(s)} \leq 1$ and $softmax(\langle \boldsymbol{q}_+^{(s)}, \boldsymbol{k}_+^{(s)} \rangle) \leq 1$. For the second inequality, we first consider $\frac{\sum_{j=2}^{M} \exp(\langle \boldsymbol{q}_+^{(s)}, \boldsymbol{k}_{n,j}^{(s)} \rangle)}{\exp(\langle \boldsymbol{q}_+^{(s)}, \boldsymbol{k}_+^{(s)} \rangle) + \sum_{j=2}^{M} \exp(\langle \boldsymbol{q}_+^{(s)}, \boldsymbol{k}_{n,j}^{(s)} \rangle)} \leq \frac{\sum_{j=2}^{M} \exp(\langle \boldsymbol{q}_+^{(s)}, \boldsymbol{k}_{n,j}^{(s)} \rangle)}{\exp(\langle \boldsymbol{q}_+^{(s)}, \boldsymbol{k}_+^{(s)} \rangle)}$, then by the monotonicity of $\langle \boldsymbol{q}_+^{(s)}, \boldsymbol{k}_{n,j}^{(s)} \rangle$ and $\langle \boldsymbol{q}_+^{(T_1)}, \boldsymbol{k}_{n,j}^{(T_1)} \rangle = o(1)$ we have $\sum_{j=2}^{M} \exp(\langle \boldsymbol{q}_+^{(s)}, \boldsymbol{k}_{n,j}^{(s)} \rangle) \leq C$ for $s \in [T_1, t]$. The last inequality is by $V_+^{(s)}, V_{n,i}^{(s)} = o(1)$ for $s \in [T_1, t]$ and absorbing the constant factors. Similarly, we have

$$\alpha_{-,-}^{(s)} \leq \frac{\eta C_9 \|\boldsymbol{\mu}\|_2^2}{\exp(\langle \boldsymbol{q}_-^{(s)}, \boldsymbol{k}_-^{(s)} \rangle)},$$

$$\beta_{+,+}^{(s)} \leq \frac{\eta C_9 \|\boldsymbol{\mu}\|_2^2}{\exp(\langle \boldsymbol{q}_+^{(s)}, \boldsymbol{k}_+^{(s)} \rangle)},$$

$$\beta_{-,-}^{(s)} \leq \frac{\eta C_9 \|\boldsymbol{\mu}\|_2^2}{\exp(\langle \boldsymbol{q}_-^{(s)}, \boldsymbol{k}_-^{(s)} \rangle)}.$$

$$\alpha_{n,+,j}^{(s)} = -\frac{\eta}{NM} \ell_n'^{(s)} \|\boldsymbol{\mu}\|_2^2$$

$$\cdot \left( -V_+^{(s)} \cdot \frac{\exp(\langle \boldsymbol{q}_+^{(s)}, \boldsymbol{k}_+^{(s)} \rangle)}{\exp(\langle \boldsymbol{q}_+^{(s)}, \boldsymbol{k}_+^{(s)} \rangle) + \sum_{j'=2}^{M} \exp(\langle \boldsymbol{q}_+^{(s)}, \boldsymbol{k}_{n,j'}^{(s)} \rangle)} \right.$$

$$\cdot \frac{\exp(\langle \boldsymbol{q}_+^{(s)}, \boldsymbol{k}_{n,j}^{(s)} \rangle)}{\exp(\langle \boldsymbol{q}_+^{(s)}, \boldsymbol{k}_+^{(s)} \rangle) + \sum_{j'=2}^{M} \exp(\langle \boldsymbol{q}_+^{(s)}, \boldsymbol{k}_{n,j'}^{(s)} \rangle)}$$

$$\left. + V_{n,i}^{(s)} \left( \frac{\exp(\langle \boldsymbol{q}_+^{(s)}, \boldsymbol{k}_{n,j}^{(s)} \rangle)}{\exp(\langle \boldsymbol{q}_+^{(s)}, \boldsymbol{k}_+^{(s)} \rangle) + \sum_{j'=2}^{M} \exp(\langle \boldsymbol{q}_+^{(s)}, \boldsymbol{k}_{n,j'}^{(s)} \rangle)} \right. \right.$$

$$- \Big( \frac{\exp(\langle \boldsymbol{q}_+^{(s)}, \boldsymbol{k}_{n,j}^{(s)} \rangle)}{\exp(\langle \boldsymbol{q}_+^{(s)}, \boldsymbol{k}_+^{(s)} \rangle) + \sum\limits_{j'=2}^{M} \exp(\langle \boldsymbol{q}_+^{(s)}, \boldsymbol{k}_{n,j'}^{(s)} \rangle)} \Big)^2 \Big)$$

$$- \sum_{k \neq j} \Big( V_{n,k}^{(s)} \cdot \frac{\exp(\langle \boldsymbol{q}_+^{(s)}, \boldsymbol{k}_{n,j}^{(s)} \rangle)}{\exp(\langle \boldsymbol{q}_+^{(s)}, \boldsymbol{k}_+^{(s)} \rangle) + \sum\limits_{j'=2}^{M} \exp(\langle \boldsymbol{q}_+^{(s)}, \boldsymbol{k}_{n,j'}^{(s)} \rangle)}$$

$$\cdot \frac{\exp(\langle \boldsymbol{q}_+^{(s)}, \boldsymbol{k}_{n,k}^{(s)} \rangle)}{\exp(\langle \boldsymbol{q}_+^{(s)}, \boldsymbol{k}_+^{(s)} \rangle) + \sum\limits_{j'=2}^{M} \exp(\langle \boldsymbol{q}_+^{(s)}, \boldsymbol{k}_{n,j'}^{(s)} \rangle)} \Big) \Big)$$

$$\geq - \frac{\eta}{NM} \ell_n'^{(s)} \|\boldsymbol{\mu}\|_2^2 \cdot \frac{\exp(\langle \boldsymbol{q}_+^{(s)}, \boldsymbol{k}_{n,j}^{(s)} \rangle)}{\exp(\langle \boldsymbol{q}_+^{(s)}, \boldsymbol{k}_+^{(s)} \rangle) + \sum\limits_{j'=2}^{M} \exp(\langle \boldsymbol{q}_+^{(s)}, \boldsymbol{k}_{n,j'}^{(s)} \rangle)}$$

$$\cdot \Big( - V_+^{(s)} \cdot \frac{\exp(\langle \boldsymbol{q}_+^{(s)}, \boldsymbol{k}_+^{(s)} \rangle)}{\exp(\langle \boldsymbol{q}_+^{(s)}, \boldsymbol{k}_+^{(s)} \rangle) + \sum\limits_{j'=2}^{M} \exp(\langle \boldsymbol{q}_+^{(s)}, \boldsymbol{k}_{n,j'}^{(s)} \rangle)}$$

$$- |V_{n,i}^{(s)}| \Big( 1 - \frac{\exp(\langle \boldsymbol{q}_+^{(s)}, \boldsymbol{k}_{n,j}^{(s)} \rangle)}{\exp(\langle \boldsymbol{q}_+^{(s)}, \boldsymbol{k}_+^{(s)} \rangle) + \sum\limits_{j'=2}^{M} \exp(\langle \boldsymbol{q}_+^{(s)}, \boldsymbol{k}_{n,j'}^{(s)} \rangle)} \Big)$$

$$- \max_l |V_{n,l}^{(s)}| \cdot \frac{\sum\limits_{k \neq j} \exp(\langle \boldsymbol{q}_+^{(s)}, \boldsymbol{k}_{n,k}^{(s)} \rangle)}{\exp(\langle \boldsymbol{q}_+^{(s)}, \boldsymbol{k}_+^{(s)} \rangle) + \sum\limits_{j'=2}^{M} \exp(\langle \boldsymbol{q}_+^{(s)}, \boldsymbol{k}_{n,j'}^{(s)} \rangle)} \Big)$$

$$\geq - \frac{2\eta}{NM} \|\boldsymbol{\mu}\|_2^2 V_+^{(s)} \cdot \frac{\exp(\langle \boldsymbol{q}_+^{(s)}, \boldsymbol{k}_{n,j}^{(s)} \rangle)}{\exp(\langle \boldsymbol{q}_+^{(s)}, \boldsymbol{k}_+^{(s)} \rangle) + \sum\limits_{j'=2}^{M} \exp(\langle \boldsymbol{q}_+^{(s)}, \boldsymbol{k}_{n,j'}^{(s)} \rangle)}$$

$$\geq - \frac{2\eta}{NM} \|\boldsymbol{\mu}\|_2^2 V_+^{(s)} \cdot \frac{\exp(\langle \boldsymbol{q}_+^{(s)}, \boldsymbol{k}_{n,j}^{(s)} \rangle)}{C}$$

$$\geq - \frac{\eta C_9 \|\boldsymbol{\mu}\|_2^2}{N} \cdot \exp(\langle \boldsymbol{q}_+^{(s)}, \boldsymbol{k}_{n,j}^{(s)} \rangle),$$

where the second inequality is by $V_+^{(s)} \geq 3M \cdot |V_{n,i}^{(s)}|$, $-\ell_n'^{(s)} \leq 1$ and the property that attention $< 1$.
For the third inequality, we consider $\frac{\exp(\langle \boldsymbol{q}_+^{(s)}, \boldsymbol{k}_{n,j}^{(s)} \rangle)}{\exp(\langle \boldsymbol{q}_+^{(s)}, \boldsymbol{k}_+^{(s)} \rangle) + \sum\limits_{j'=2}^{M} \exp(\langle \boldsymbol{q}_+^{(s)}, \boldsymbol{k}_{n,j'}^{(s)} \rangle)} \leq \frac{\exp(\langle \boldsymbol{q}_+^{(s)}, \boldsymbol{k}_{n,j}^{(s)} \rangle)}{\exp(\langle \boldsymbol{q}_+^{(s)}, \boldsymbol{k}_+^{(s)} \rangle)}$ first, then
by the monotonicity of $\langle \boldsymbol{q}_+^{(s)}, \boldsymbol{k}_+^{(s)} \rangle$ and $\langle \boldsymbol{q}_+^{(T_1)}, \boldsymbol{k}_+^{(T_1)} \rangle$ we have $\exp(\langle \boldsymbol{q}_+^{(s)}, \boldsymbol{k}_+^{(s)} \rangle) \geq C$ for $s \in [T_1, t]$.
The last inequality is by $V_+^{(s)} = o(1)$ for $s \in [T_1, t]$ and absorbing the constant factors. Similarly, we
have

$$\alpha_{n,-,j}^{(s)} \geq - \frac{\eta C_9 \|\boldsymbol{\mu}\|_2^2}{N} \cdot \exp(\langle \boldsymbol{q}_-^{(s)}, \boldsymbol{k}_{n,j}^{(s)} \rangle).$$

$$\beta_{n,+,i}^{(s)} = - \frac{\eta \|\boldsymbol{\mu}\|_2^2}{NM} \ell_n'^{(s)}$$

$$\cdot \Big( V_+^{(s)} \Big( \frac{\exp(\langle \boldsymbol{q}_{n,i}^{(s)}, \boldsymbol{k}_+^{(s)} \rangle)}{\exp(\langle \boldsymbol{q}_{n,i}^{(s)}, \boldsymbol{k}_+^{(s)} \rangle) + \sum\limits_{j=2}^{M} \exp(\langle \boldsymbol{q}_{n,i}^{(s)}, \boldsymbol{k}_{n,j}^{(s)} \rangle)}$$

$$- \left( \frac{\exp(\langle \boldsymbol{q}_{n,i}^{(s)}, \boldsymbol{k}_+^{(s)} \rangle)}{\exp(\langle \boldsymbol{q}_{n,i}^{(s)}, \boldsymbol{k}_+^{(s)} \rangle) + \sum\limits_{j=2}^{M} \exp(\langle \boldsymbol{q}_{n,i}^{(s)}, \boldsymbol{k}_{n,j}^{(s)} \rangle)} \right)^2 \Big)$$

$$- \sum_{k=2}^{M} \Big( V_{n,i}^{(s)} \cdot \frac{\exp(\langle \boldsymbol{q}_{n,i}^{(s)}, \boldsymbol{k}_+^{(s)} \rangle)}{\exp(\langle \boldsymbol{q}_{n,i}^{(s)}, \boldsymbol{k}_+^{(s)} \rangle) + \sum\limits_{j=2}^{M} \exp(\langle \boldsymbol{q}_{n,i}^{(s)}, \boldsymbol{k}_{n,j}^{(s)} \rangle)}$$

$$\cdot \frac{\exp(\langle \boldsymbol{q}_{n,i}^{(s)}, \boldsymbol{k}_{n,k}^{(s)} \rangle)}{\exp(\langle \boldsymbol{q}_{n,i}^{(s)}, \boldsymbol{k}_+^{(s)} \rangle) + \sum\limits_{j=2}^{M} \exp(\langle \boldsymbol{q}_{n,i}^{(s)}, \boldsymbol{k}_{n,j}^{(s)} \rangle)} \Big) \Big)$$

$$= -\frac{\eta \|\boldsymbol{\mu}\|_2^2}{NM} \ell_n'^{(s)} \frac{\exp(\langle \boldsymbol{q}_{n,i}^{(s)}, \boldsymbol{k}_+^{(s)} \rangle)}{\exp(\langle \boldsymbol{q}_{n,i}^{(s)}, \boldsymbol{k}_+^{(s)} \rangle) + \sum\limits_{j=2}^{M} \exp(\langle \boldsymbol{q}_{n,i}^{(s)}, \boldsymbol{k}_{n,j}^{(s)} \rangle)}$$

$$\cdot \Big( V_+^{(s)} \big( 1 - \frac{\exp(\langle \boldsymbol{q}_{n,i}^{(s)}, \boldsymbol{k}_+^{(s)} \rangle)}{\exp(\langle \boldsymbol{q}_{n,i}^{(s)}, \boldsymbol{k}_+^{(s)} \rangle) + \sum\limits_{j=2}^{M} \exp(\langle \boldsymbol{q}_{n,i}^{(s)}, \boldsymbol{k}_{n,j}^{(s)} \rangle)} \big)$$

$$- \sum_{k=2}^{M} \Big( V_{n,i}^{(s)} \cdot \frac{\exp(\langle \boldsymbol{q}_{n,i}^{(s)}, \boldsymbol{k}_{n,k}^{(s)} \rangle)}{\exp(\langle \boldsymbol{q}_{n,i}^{(s)}, \boldsymbol{k}_+^{(s)} \rangle) + \sum\limits_{j=2}^{M} \exp(\langle \boldsymbol{q}_{n,i}^{(s)}, \boldsymbol{k}_{n,j}^{(s)} \rangle)} \Big) \Big)$$

$$\leq -\frac{\eta \|\boldsymbol{\mu}\|_2^2}{NM} \ell_n'^{(s)} \frac{\exp(\langle \boldsymbol{q}_{n,i}^{(s)}, \boldsymbol{k}_+^{(s)} \rangle)}{\exp(\langle \boldsymbol{q}_{n,i}^{(s)}, \boldsymbol{k}_+^{(s)} \rangle) + \sum\limits_{j=2}^{M} \exp(\langle \boldsymbol{q}_{n,i}^{(s)}, \boldsymbol{k}_{n,j}^{(s)} \rangle)}$$

$$\cdot \Big( V_+^{(s)} \big( 1 - \frac{\exp(\langle \boldsymbol{q}_{n,i}^{(s)}, \boldsymbol{k}_+^{(s)} \rangle)}{\exp(\langle \boldsymbol{q}_{n,i}^{(s)}, \boldsymbol{k}_+^{(s)} \rangle) + \sum\limits_{j=2}^{M} \exp(\langle \boldsymbol{q}_{n,i}^{(s)}, \boldsymbol{k}_{n,j}^{(s)} \rangle)} \big)$$

$$+ |V_{n,i}^{(s)}| \big( \frac{\sum\limits_{k=2}^{M} \exp(\langle \boldsymbol{q}_{n,i}^{(s)}, \boldsymbol{k}_{n,k}^{(s)} \rangle)}{\exp(\langle \boldsymbol{q}_{n,i}^{(s)}, \boldsymbol{k}_+^{(s)} \rangle) + \sum\limits_{j=2}^{M} \exp(\langle \boldsymbol{q}_{n,i}^{(s)}, \boldsymbol{k}_{n,j}^{(s)} \rangle)} \big) \Big)$$

$$\leq -\frac{\eta \|\boldsymbol{\mu}\|_2^2}{NM} \ell_n'^{(s)}$$

$$\cdot \Big( V_+^{(s)} \cdot \frac{\sum\limits_{j=2}^{M} \exp(\langle \boldsymbol{q}_{n,i}^{(s)}, \boldsymbol{k}_{n,j}^{(s)} \rangle)}{\exp(\langle \boldsymbol{q}_{n,i}^{(s)}, \boldsymbol{k}_+^{(s)} \rangle) + \sum\limits_{j=2}^{M} \exp(\langle \boldsymbol{q}_{n,i}^{(s)}, \boldsymbol{k}_{n,j}^{(s)} \rangle)}$$

$$+ |V_{n,i}^{(s)}| \cdot \frac{\sum\limits_{k=2}^{M} \exp(\langle \boldsymbol{q}_{n,i}^{(s)}, \boldsymbol{k}_{n,k}^{(s)} \rangle)}{\exp(\langle \boldsymbol{q}_{n,i}^{(s)}, \boldsymbol{k}_+^{(s)} \rangle) + \sum\limits_{j=2}^{M} \exp(\langle \boldsymbol{q}_{n,i}^{(s)}, \boldsymbol{k}_{n,j}^{(s)} \rangle)} \Big)$$

$$\leq -\frac{\eta \|\boldsymbol{\mu}\|_2^2}{NM} \ell_n'^{(s)} \cdot \Big( V_+^{(s)} \cdot \frac{C}{\exp(\langle \boldsymbol{q}_{n,i}^{(s)}, \boldsymbol{k}_+^{(s)} \rangle)} + |V_{n,i}^{(s)}| \cdot \frac{C}{\exp(\langle \boldsymbol{q}_{n,i}^{(s)}, \boldsymbol{k}_+^{(s)} \rangle)} \Big)$$

$$\leq \frac{\eta C_9 \|\boldsymbol{\mu}\|_2^2}{N \exp(\langle \boldsymbol{q}_{n,i}^{(s)}, \boldsymbol{k}_+^{(s)} \rangle)},$$

where the second inequality is by $softmax(\langle \boldsymbol{q}_{n,i}^{(s)}, \boldsymbol{k}_+^{(s)} \rangle) \leq 1$. For the second inequality, we

first consider $\dfrac{\sum\limits_{j=2}^{M} \exp(\langle \boldsymbol{q}_{n,i}^{(s)}, \boldsymbol{k}_{n,j}^{(s)} \rangle)}{\exp(\langle \boldsymbol{q}_{n,i}^{(s)}, \boldsymbol{k}_+^{(s)} \rangle) + \sum\limits_{j=2}^{M} \exp(\langle \boldsymbol{q}_{n,i}^{(s)}, \boldsymbol{k}_{n,j}^{(s)} \rangle)} \leq \dfrac{\sum\limits_{j=2}^{M} \exp(\langle \boldsymbol{q}_{n,i}^{(s)}, \boldsymbol{k}_{n,j}^{(s)} \rangle)}{\exp(\langle \boldsymbol{q}_{n,i}^{(s)}, \boldsymbol{k}_+^{(s)} \rangle)}$, then by the monotonicity of

$\langle \boldsymbol{q}_{n,i}^{(s)}, \boldsymbol{k}_{n,j}^{(s)} \rangle$ and $\langle \boldsymbol{q}_{n,i}^{(T_1)}, \boldsymbol{k}_{n,j}^{(T_1)} \rangle = o(1)$ we have $\sum\limits_{j=2}^{M} \exp(\langle \boldsymbol{q}_{n,i}^{(s)}, \boldsymbol{k}_{n,j}^{(s)} \rangle) \leq C$ for $s \in [T_1, t]$. The last

inequality is by $V_+^{(s)}, V_{n,i}^{(s)} = o(1)$ for $s \in [T_1, t]$ and absorbing the constant factors. Similarly, we have

$$\beta_{n,-,i}^{(s)} \leq \frac{\eta C_9 \|\boldsymbol{\mu}\|_2^2}{N \exp(\langle \boldsymbol{q}_{n,i}^{(s)}, \boldsymbol{k}_-^{(s)} \rangle)},$$

$$\alpha_{n,i,+}^{(s)} = \frac{\eta \|\boldsymbol{\xi}_{n,i}\|_2^2}{NM} - \ell_n'^{(s)}$$

$$\cdot \left( V_+^{(s)} \left( \frac{\exp(\langle \boldsymbol{q}_{n,i}^{(s)}, \boldsymbol{k}_+^{(s)} \rangle)}{\exp(\langle \boldsymbol{q}_{n,i}^{(s)}, \boldsymbol{k}_+^{(s)} \rangle) + \sum\limits_{j=2}^{M} \exp(\langle \boldsymbol{q}_{n,i}^{(s)}, \boldsymbol{k}_{n,j}^{(s)} \rangle)} \right. \right.$$

$$- \left( \frac{\exp(\langle \boldsymbol{q}_{n,i}^{(s)}, \boldsymbol{k}_+^{(s)} \rangle)}{\exp(\langle \boldsymbol{q}_{n,i}^{(s)}, \boldsymbol{k}_+^{(s)} \rangle) + \sum\limits_{j=2}^{M} \exp(\langle \boldsymbol{q}_{n,i}^{(s)}, \boldsymbol{k}_{n,j}^{(s)} \rangle)} \right)^2 \right)$$

$$- \sum\limits_{k=2}^{M} \left( V_{n,i}^{(s)} \cdot \frac{\exp(\langle \boldsymbol{q}_{n,i}^{(s)}, \boldsymbol{k}_+^{(s)} \rangle)}{\exp(\langle \boldsymbol{q}_{n,i}^{(s)}, \boldsymbol{k}_+^{(s)} \rangle) + \sum\limits_{j=2}^{M} \exp(\langle \boldsymbol{q}_{n,i}^{(s)}, \boldsymbol{k}_{n,j}^{(s)} \rangle)} \right.$$

$$\left. \left. \cdot \frac{\exp(\langle \boldsymbol{q}_{n,i}^{(s)}, \boldsymbol{k}_{n,k}^{(s)} \rangle)}{\exp(\langle \boldsymbol{q}_{n,i}^{(s)}, \boldsymbol{k}_+^{(s)} \rangle) + \sum\limits_{j=2}^{M} \exp(\langle \boldsymbol{q}_{n,i}^{(s)}, \boldsymbol{k}_{n,j}^{(s)} \rangle)} \right) \right)$$

$$+ \{lower\ order\ term\}$$

$$= \frac{\eta \|\boldsymbol{\xi}_{n,i}\|_2^2}{NM} - \ell_n'^{(s)} \frac{\exp(\langle \boldsymbol{q}_{n,i}^{(s)}, \boldsymbol{k}_+^{(s)} \rangle)}{\exp(\langle \boldsymbol{q}_{n,i}^{(s)}, \boldsymbol{k}_+^{(s)} \rangle) + \sum\limits_{j=2}^{M} \exp(\langle \boldsymbol{q}_{n,i}^{(s)}, \boldsymbol{k}_{n,j}^{(s)} \rangle)}$$

$$\cdot \left( V_+^{(s)} \left( 1 - \frac{\exp(\langle \boldsymbol{q}_{n,i}^{(s)}, \boldsymbol{k}_+^{(s)} \rangle)}{\exp(\langle \boldsymbol{q}_{n,i}^{(s)}, \boldsymbol{k}_+^{(s)} \rangle) + \sum\limits_{j=2}^{M} \exp(\langle \boldsymbol{q}_{n,i}^{(s)}, \boldsymbol{k}_{n,j}^{(s)} \rangle)} \right) \right.$$

$$\left. - \sum\limits_{k=2}^{M} \left( V_{n,i}^{(s)} \cdot \frac{\exp(\langle \boldsymbol{q}_{n,i}^{(s)}, \boldsymbol{k}_{n,k}^{(s)} \rangle)}{\exp(\langle \boldsymbol{q}_{n,i}^{(s)}, \boldsymbol{k}_+^{(s)} \rangle) + \sum\limits_{j=2}^{M} \exp(\langle \boldsymbol{q}_{n,i}^{(s)}, \boldsymbol{k}_{n,j}^{(s)} \rangle)} \right) \right)$$

$$+ \{lower\ order\ term\}$$

$$\leq \frac{\eta \|\boldsymbol{\xi}_{n,i}\|_2^2}{NM}$$

$$\cdot \left( V_+^{(s)} \left( 1 - \frac{\exp(\langle \boldsymbol{q}_{n,i}^{(s)}, \boldsymbol{k}_+^{(s)} \rangle)}{\exp(\langle \boldsymbol{q}_{n,i}^{(s)}, \boldsymbol{k}_+^{(s)} \rangle) + \sum\limits_{j=2}^{M} \exp(\langle \boldsymbol{q}_{n,i}^{(s)}, \boldsymbol{k}_{n,j}^{(s)} \rangle)} \right) \right.$$

$$- \sum_{k=2}^{M} \Big( V_{n,i}^{(s)} \cdot \frac{\exp(\langle \boldsymbol{q}_{n,i}^{(s)}, \boldsymbol{k}_{n,k}^{(s)}\rangle)}{\exp(\langle \boldsymbol{q}_{n,i}^{(s)}, \boldsymbol{k}_{+}^{(s)}\rangle) + \sum\limits_{j=2}^{M} \exp(\langle \boldsymbol{q}_{n,i}^{(s)}, \boldsymbol{k}_{n,j}^{(s)}\rangle)} \Big) \Big)$$

$$+ \{lower\ order\ term\}$$

$$\leq \frac{\eta \|\boldsymbol{\xi}_{n,i}\|_2^2}{NM} \cdot \Big( V_{+}^{(s)} \cdot \frac{\sum\limits_{j=2}^{M} \exp(\langle \boldsymbol{q}_{n,i}^{(s)}, \boldsymbol{k}_{n,j}^{(s)}\rangle)}{\exp(\langle \boldsymbol{q}_{n,i}^{(s)}, \boldsymbol{k}_{+}^{(s)}\rangle) + \sum\limits_{j=2}^{M} \exp(\langle \boldsymbol{q}_{n,i}^{(s)}, \boldsymbol{k}_{n,j}^{(s)}\rangle)}$$

$$+ |V_{n,i}^{(s)}| \cdot \frac{\sum\limits_{k=2}^{M} \exp(\langle \boldsymbol{q}_{n,i}^{(s)}, \boldsymbol{k}_{n,k}^{(s)}\rangle)}{\exp(\langle \boldsymbol{q}_{n,i}^{(s)}, \boldsymbol{k}_{+}^{(s)}\rangle) + \sum\limits_{j=2}^{M} \exp(\langle \boldsymbol{q}_{n,i}^{(s)}, \boldsymbol{k}_{n,j}^{(s)}\rangle)} \Big)$$

$$+ \{lower\ order\ term\}$$

$$\leq \frac{\eta \|\boldsymbol{\xi}_{n,i}\|_2^2}{NM} \cdot \Big( V_{+}^{(s)} \cdot \frac{C}{\exp(\langle \boldsymbol{q}_{n,i}^{(s)}, \boldsymbol{k}_{+}^{(s)}\rangle)} + |V_{n,i}^{(s)}| \cdot \frac{C}{\exp(\langle \boldsymbol{q}_{n,i}^{(s)}, \boldsymbol{k}_{+}^{(s)}\rangle)} \Big)$$

$$+ \{lower\ order\ term\}$$

$$\leq \frac{\eta C_9 \sigma_p^2 d}{N \exp(\langle \boldsymbol{q}_{n,i}^{(s)}, \boldsymbol{k}_{+}^{(s)}\rangle)},$$

where most of these processes are similar to the other equations above, and the last inequality we absorb the constant factors and the $\{lower\ order\ term\}$. Similarly, we have

$$\alpha_{n,i,-}^{(s)} \leq \frac{\eta C_9 \sigma_p^2 d}{N \exp(\langle \boldsymbol{q}_{n,i}^{(s)}, \boldsymbol{k}_{-}^{(s)}\rangle)},$$

$$\beta_{n,i,+}^{(s)} \geq -\frac{\eta C_9 \sigma_p^2 d}{N} \exp(\langle \boldsymbol{q}_{+}^{(s)}, \boldsymbol{k}_{n,i}^{(s)}\rangle),$$

$$\beta_{n,i,-}^{(s)} \geq -\frac{\eta C_9 \sigma_p^2 d}{N} \exp(\langle \boldsymbol{q}_{-}^{(s)}, \boldsymbol{k}_{n,i}^{(s)}\rangle),$$

$$\alpha_{n,i,n,j}^{(s)} \geq -\frac{\eta C_9 \sigma_p^2 d}{N} \exp(\langle \boldsymbol{q}_{n,i}^{(s)}, \boldsymbol{k}_{n,j}^{(s)}\rangle),$$

$$\beta_{n,j,n,i}^{(s)} \geq -\frac{\eta C_9 \sigma_p^2 d}{N} \exp(\langle \boldsymbol{q}_{n,i}^{(s)}, \boldsymbol{k}_{n,j}^{(s)}\rangle),$$

Similar to F.5, we apply the bounds of $\alpha$ and $\beta$ above to give the upper bounds for the dynamics $\langle \boldsymbol{q}, \boldsymbol{k} \rangle$.

$$\langle \boldsymbol{q}_+^{(s+1)}, \boldsymbol{k}_+^{(s+1)}\rangle - \langle \boldsymbol{q}_+^{(s)}, \boldsymbol{k}_+^{(s)}\rangle$$

$$= \alpha_{+,+}^{(s)}\|\boldsymbol{k}_+^{(s)}\|_2^2 + \sum_{n\in S_+}\sum_{i=2}^M \alpha_{n,+,i}^{(s)}\langle \boldsymbol{k}_+^{(s)}, \boldsymbol{k}_{n,i}^{(s)}\rangle$$

$$+ \beta_{+,+}^{(s)}\|\boldsymbol{q}_+^{(s)}\|_2^2 + \sum_{n\in S_+}\sum_{i=2}^M \beta_{n,+,i}^{(s)}\langle \boldsymbol{q}_+^{(s)}, \boldsymbol{q}_{n,i}^{(s)}\rangle$$

$$+ \left( \alpha_{+,+}^{(s)}\boldsymbol{k}_+^{(s)} + \sum_{n\in S_+}\sum_{i=2}^M \alpha_{n,+,i}^{(s)}\boldsymbol{k}_{n,i}^{(s)}\right) \tag{243}$$

$$\cdot \left( \beta_{+,+}^{(s)}\boldsymbol{q}_+^{(s)\top} + \sum_{n\in S_+}\sum_{i=2}^M \beta_{n,+,i}^{(s)}\boldsymbol{q}_{n,i}^{(s)\top}\right)$$

$$= \alpha_{+,+}^{(s)}\|\boldsymbol{k}_+^{(s)}\|_2^2 + \beta_{+,+}^{(s)}\|\boldsymbol{q}_+^{(s)}\|_2^2 + \{lower\ order\ term\}$$

$$\leq \frac{2\eta C_9\|\boldsymbol{\mu}\|_2^2}{\exp(\langle \boldsymbol{q}_+^{(s)}, \boldsymbol{k}_+^{(s)}\rangle)}\cdot \Theta(\|\boldsymbol{\mu}\|_2^2\sigma_h^2 d_h) + \{lower\ order\ term\}$$

$$\leq \frac{\eta C_{10}\|\boldsymbol{\mu}\|_2^4\sigma_h^2 d_h}{\exp(\langle \boldsymbol{q}_+^{(s)}, \boldsymbol{k}_+^{(s)}\rangle)},$$

similarly, we have

$$\langle \boldsymbol{q}_-^{(s+1)}, \boldsymbol{k}_-^{(s+1)}\rangle - \langle \boldsymbol{q}_-^{(s)}, \boldsymbol{k}_-^{(s)}\rangle \leq \frac{\eta C_{10}\|\boldsymbol{\mu}\|_2^4\sigma_h^2 d_h}{\exp(\langle \boldsymbol{q}_-^{(s)}, \boldsymbol{k}_-^{(s)}\rangle)}. \tag{244}$$

$$\langle \boldsymbol{q}_+^{(s+1)}, \boldsymbol{k}_{n,j}^{(s+1)}\rangle - \langle \boldsymbol{q}_+^{(s)}, \boldsymbol{k}_{n,j}^{(s)}\rangle$$

$$= \alpha_{+,+}^{(s)}\langle \boldsymbol{k}_+^{(s)}, \boldsymbol{k}_{n,j}^{(s)}\rangle + \sum_{n'\in S_+}\sum_{l=2}^M \alpha_{n',+,l}^{(s)}\langle \boldsymbol{k}_{n,j}^{(s)}, \boldsymbol{k}_{n',l}^{(s)}\rangle$$

$$+ \beta_{n,j,+}^{(s)}\|\boldsymbol{q}_+^{(s)}\|_2^2 + \beta_{n,j,-}^{(s)}\langle \boldsymbol{q}_+^{(s)}, \boldsymbol{q}_-^{(s)}\rangle + \sum_{n'=1}^N\sum_{l=2}^M \beta_{n,j,n',l}^{(s)}\langle \boldsymbol{q}_+^{(s)}, \boldsymbol{q}_{n',l}^{(s)}\rangle$$

$$+ \left( \alpha_{+,+}^{(s)}\boldsymbol{k}_+^{(s)} + \sum_{n'\in S_+}\sum_{l=2}^M \alpha_{n',+,l}^{(s)}\boldsymbol{k}_{n',l}^{(s)}\right) \tag{245}$$

$$\cdot \left( \beta_{n,j,+}^{(s)}\boldsymbol{q}_+^{(s)\top} + \beta_{n,j,-}^{(s)}\boldsymbol{q}_-^{(s)\top} + \sum_{n'=1}^N\sum_{l=2}^M \beta_{n,j,n',l}^{(s)}\boldsymbol{q}_{n',l}^{(s)\top}\right)$$

$$= \alpha_{n,+,j}^{(s)}\|\boldsymbol{k}_{n,j}^{(s)}\|_2^2 + \beta_{n,j,+}^{(s)}\|\boldsymbol{q}_+^{(s)}\|_2^2 + \{lower\ order\ term\}$$

$$\geq -\frac{\eta C_9\|\boldsymbol{\mu}\|_2^2}{N}\cdot \exp(\langle \boldsymbol{q}_+^{(s)}, \boldsymbol{k}_{n,j}^{(s)}\rangle)\cdot \Theta\left( \sigma_p^2\sigma_h^2 dd_h\right)$$

$$- \frac{\eta C_9\sigma_p^2 d}{N}\exp(\langle \boldsymbol{q}_+^{(s)}, \boldsymbol{k}_{n,j}^{(s)}\rangle)\cdot \Theta\left( \|\boldsymbol{\mu}\|_2^2\sigma_h^2 d_h\right)$$

$$+ \{lower\ order\ term\}$$

$$\geq -\frac{\eta C_{10}\sigma_p^2 d\|\boldsymbol{\mu}\|_2^2\sigma_h^2 d_h}{N}\cdot \exp(\langle \boldsymbol{q}_+^{(s)}, \boldsymbol{k}_{n,j}^{(s)}\rangle),$$

similarly, we have

$$\langle \boldsymbol{q}_-^{(s+1)}, \boldsymbol{k}_{n,j}^{(s+1)}\rangle - \langle \boldsymbol{q}_-^{(s)}, \boldsymbol{k}_{n,j}^{(s)}\rangle \geq -\frac{\eta C_{10}\sigma_p^2 d\|\boldsymbol{\mu}\|_2^2\sigma_h^2 d_h}{N}\cdot \exp(\langle \boldsymbol{q}_-^{(s)}, \boldsymbol{k}_{n,j}^{(s)}\rangle). \tag{246}$$

$$\langle \boldsymbol{q}_{n,i}^{(s+1)}, \boldsymbol{k}_+^{(s+1)} \rangle - \langle \boldsymbol{q}_{n,i}^{(s)}, \boldsymbol{k}_+^{(s)} \rangle$$

$$= \alpha_{n,i,+}^{(s)} \|\boldsymbol{k}_+^{(s)}\|_2^2 + \alpha_{n,i,-}^{(s)} \langle \boldsymbol{k}_+^{(s)}, \boldsymbol{k}_-^{(s)} \rangle + \sum_{n'=1}^{N} \sum_{l=2}^{M} \alpha_{n,i,n',l}^{(s)} \langle \boldsymbol{k}_+^{(s)}, \boldsymbol{k}_{n',l}^{(s)} \rangle$$

$$+ \beta_{+,+}^{(s)} \langle \boldsymbol{q}_+^{(s)}, \boldsymbol{q}_{n,i}^{(s)} \rangle + \sum_{n' \in S_+} \sum_{l=2}^{M} \beta_{n',+,l}^{(s)} \langle \boldsymbol{q}_{n,i}^{(s)}, \boldsymbol{q}_{n',l}^{(s)} \rangle$$

$$+ \left( \alpha_{n,i,+}^{(s)} \boldsymbol{k}_+^{(s)} + \alpha_{n,i,-}^{(s)} \boldsymbol{k}_-^{(s)} + \sum_{n'=1}^{N} \sum_{l=2}^{M} \alpha_{n,i,n',l}^{(s)} \boldsymbol{k}_{n',l}^{(s)} \right)$$

$$\cdot \left( \beta_{+,+}^{(s)} \boldsymbol{q}_+^{(s)\top} + \sum_{n' \in S_+} \sum_{l=2}^{M} \beta_{n',+,l}^{(s)} \boldsymbol{q}_{n',l}^{(s)\top} \right) \tag{247}$$

$$= \alpha_{n,i,+}^{(s)} \|\boldsymbol{k}_+^{(s)}\|_2^2 + \beta_{n,+,i}^{(s)} \|\boldsymbol{q}_{n,i}^{(s)}\|_2^2 + \{lower\ order\ term\}$$

$$\leq \frac{\eta C_9 \sigma_p^2 d}{N \exp(\langle \boldsymbol{q}_{n,i}^{(s)}, \boldsymbol{k}_+^{(s)} \rangle)} \cdot \Theta\left( \|\boldsymbol{\mu}\|_2^2 \sigma_h^2 d_h \right)$$

$$+ \frac{\eta C_9 \|\boldsymbol{\mu}\|_2^2}{N \exp(\langle \boldsymbol{q}_{n,i}^{(s)}, \boldsymbol{k}_+^{(s)} \rangle)} \cdot \Theta\left( \sigma_p^2 \sigma_h^2 d d_h \right)$$

$$+ \{lower\ order\ term\}$$

$$\leq \frac{\eta C_{10} \sigma_p^2 d \|\boldsymbol{\mu}\|_2^2 \sigma_h^2 d_h}{N \exp(\langle \boldsymbol{q}_{n,i}^{(s)}, \boldsymbol{k}_+^{(s)} \rangle)},$$

similarly, we have

$$\langle \boldsymbol{q}_{n,i}^{(s+1)}, \boldsymbol{k}_-^{(s+1)} \rangle - \langle \boldsymbol{q}_{n,i}^{(s)}, \boldsymbol{k}_-^{(s)} \rangle \leq \frac{\eta C_{10} \sigma_p^2 d \|\boldsymbol{\mu}\|_2^2 \sigma_h^2 d_h}{N \exp(\langle \boldsymbol{q}_{n,i}^{(s)}, \boldsymbol{k}_-^{(s)} \rangle)}. \tag{248}$$

$$\langle \boldsymbol{q}_{n,i}^{(s+1)}, \boldsymbol{k}_{n,j}^{(s+1)} \rangle - \langle \boldsymbol{q}_{n,i}^{(s)}, \boldsymbol{k}_{n,j}^{(s)} \rangle$$

$$= \alpha_{n,i,+}^{(s)} \langle \boldsymbol{k}_+^{(s)}, \boldsymbol{k}_{n,j}^{(s)} \rangle + \alpha_{n,i,-}^{(s)} \langle \boldsymbol{k}_-^{(s)}, \boldsymbol{k}_{n,j}^{(s)} \rangle + \sum_{n'=1}^{N} \sum_{l=2}^{M} \alpha_{n,i,n',l}^{(s)} \langle \boldsymbol{k}_{n',l}^{(s)}, \boldsymbol{k}_{n,j}^{(s)} \rangle$$

$$+ \beta_{n,j,+}^{(s)} \langle \boldsymbol{q}_+^{(s)}, \boldsymbol{q}_{n,i}^{(s)} \rangle + \beta_{n,j,-}^{(s)} \langle \boldsymbol{q}_-^{(s)}, \boldsymbol{q}_{n,i}^{(s)} \rangle + \sum_{n'=1}^{N} \sum_{l=2}^{M} \beta_{n,j,n',l}^{(s)} \langle \boldsymbol{q}_{n',l}^{(s)}, \boldsymbol{q}_{n,i}^{(s)} \rangle$$

$$+ \left( \alpha_{n,i,+}^{(s)} \boldsymbol{k}_+^{(s)} + \alpha_{n,i,-}^{(s)} \boldsymbol{k}_-^{(s)} + \sum_{n'=1}^{N} \sum_{l=2}^{M} \alpha_{n,i,n',l}^{(s)} \boldsymbol{k}_{n',l}^{(s)} \right)$$

$$\cdot \left( \beta_{n,j,+}^{(s)} \boldsymbol{q}_+^{(s)\top} + \beta_{n,j,-}^{(s)} \boldsymbol{q}_-^{(s)\top} + \sum_{n'=1}^{N} \sum_{l=2}^{M} \beta_{n,j,n',l}^{(s)} \boldsymbol{q}_{n',l}^{(s)\top} \right) \tag{249}$$

$$= \alpha_{n,i,n,j}^{(s)} \|\boldsymbol{k}_{n,j}^{(s)}\|_2^2 + \beta_{n,j,n,i}^{(s)} \|\boldsymbol{q}_{n,i}^{(s)}\|_2^2$$

$$+ \{lower\ order\ term\}$$

$$\geq -\frac{2\eta C_9 \sigma_p^2 d}{N} \exp(\langle \boldsymbol{q}_{n,i}^{(s)}, \boldsymbol{k}_{n,j}^{(s)} \rangle) \cdot \Theta\left( \sigma_p^2 \sigma_h^2 d d_h \right)$$

$$+ \{lower\ order\ term\}$$

$$\geq -\frac{\eta C_{10} \sigma_p^4 d^2 \sigma_h^2 d_h}{N} \cdot \exp(\langle \boldsymbol{q}_{n,i}^{(s)}, \boldsymbol{k}_{n,j}^{(s)} \rangle).$$

### F.7 Bounds for the Sum of $\alpha$ and $\beta$

The gradients of the inner products of $\boldsymbol{q}$ and $\boldsymbol{k}$ contain a lot of coefficients $\alpha$ and $\beta$, and in order to conveniently give the upper bounds of some lower order inner products, we will give upper bounds for the summation of $\alpha$ and $\beta$ (e.g. $\sum_{s=T_1}^{t} |\alpha_{+,+}^{(s)}|$).

Note that in the Jacobi matrix of the Softmax function, the elements on the diagonal are $softmax(a_i) \cdot \left(1 - softmax(a_i)\right)$ and the elements on the off-diagonal are $softmax(a_i) \cdot softmax(a_j)$. In Stage II, the attentions on signals $\boldsymbol{\mu}_\pm$ increase and the attentions on noises $\boldsymbol{\xi}$ decrease, then we can consider the following cases

- if $a_i = \langle \boldsymbol{q}_+, \boldsymbol{k}_+ \rangle$ or $a_i = \langle \boldsymbol{q}_i, \boldsymbol{k}_+ \rangle$, $softmax(a_i)$ has a constant upper bound 1, $\left(1 - softmax(a_i)\right)$ decreases as $softmax(a_i)$ increases. So the upper bound of $softmax(a_i) \cdot \left(1 - softmax(a_i)\right)$ decreases as $softmax(a_i)$ increases.

- if $a_i = \langle \boldsymbol{q}_+, \boldsymbol{k}_j \rangle$ or $a_i = \langle \boldsymbol{q}_i, \boldsymbol{k}_j \rangle$, $\left(1 - softmax(a_i)\right)$ has a constant upper bound 1. So the upper bound of $softmax(a_i) \cdot \left(1 - softmax(a_i)\right)$ decreases as $softmax(a_i)$ decreases.

- if $a_j = \langle \boldsymbol{q}_+, \boldsymbol{k}_j \rangle$ or $a_j = \langle \boldsymbol{q}_i, \boldsymbol{k}_j \rangle$, $softmax(a_i)$ has a constant upper bound 1. So the upper bound of $softmax(a_i) \cdot softmax(a_j)$ decreases as $softmax(a_j)$ decreases.

Based on the above cases, we first study the bounds of the following terms

- $1 - softmax(\langle \boldsymbol{q}_+^{(s)}, \boldsymbol{k}_+^{(s)} \rangle)$

- $1 - softmax(\langle \boldsymbol{q}_{n,i}^{(s)}, \boldsymbol{k}_+^{(s)} \rangle)$

- $softmax(\langle \boldsymbol{q}_+^{(s)}, \boldsymbol{k}_{n,j}^{(s)} \rangle)$

- $softmax(\langle \boldsymbol{q}_{n,i}^{(s)}, \boldsymbol{k}_{n,j}^{(s)} \rangle)$

Note that $1 - softmax(\langle \boldsymbol{q}_+^{(s)}, \boldsymbol{k}_+^{(s)} \rangle) = \sum_j softmax(\langle \boldsymbol{q}_+^{(s)}, \boldsymbol{k}_{n,j}^{(s)} \rangle)$ and $1 - softmax(\langle \boldsymbol{q}_{n,i}^{(s)}, \boldsymbol{k}_+^{(s)} \rangle) = \sum_j softmax(\langle \boldsymbol{q}_{n,i}^{(s)}, \boldsymbol{k}_{n,j}^{(s)} \rangle)$, we only need to give the upper bounds for $softmax(\langle \boldsymbol{q}_+^{(s)}, \boldsymbol{k}_{n,j}^{(s)} \rangle)$ and $softmax(\langle \boldsymbol{q}_{n,i}^{(s)}, \boldsymbol{k}_{n,j}^{(s)} \rangle)$.

Assume that the propositions $\mathcal{B}(T_1), \ldots, \mathcal{B}(s), \mathcal{D}(T_1), \ldots, \mathcal{D}(s-1)$ hold ($s \in [T_1, t]$), we have

$$|V_\pm^{(s)}|, |V_{n,i}^{(s)}| \le O(d_h^{-\frac{1}{4}}) + \eta C_4 \|\boldsymbol{\mu}\|_2^2 \|\boldsymbol{w}_O\|_2^2 (s - T_1), \tag{250}$$

$$\Lambda_{n,\pm,j}^{(s)} \ge \log\left( \exp(\Lambda_{n,\pm,j}^{(T_1)}) + \frac{\eta^2 C_8 \|\boldsymbol{\mu}\|_2^4 \|\boldsymbol{w}_O\|_2^2 d_h^{\frac{1}{2}}}{N\left(\log(6N^2 M^2/\delta)\right)^2} \cdot (s - T_1)(s - T_1 - 1) \right), \tag{251}$$

$$\Lambda_{n,i,\pm,j}^{(s)} \ge \log\left( \exp(\Lambda_{n,i,\pm,j}^{(T_1)}) + \frac{\eta^2 C_8 \sigma_p^2 d \|\boldsymbol{\mu}\|_2^2 \|\boldsymbol{w}_O\|_2^2 d_h^{\frac{1}{2}}}{N\left(\log(6N^2 M^2/\delta)\right)^2} \cdot (s - T_1)(s - T_1 - 1) \right), \tag{252}$$

for $i, j \in [M] \backslash \{1\}, n \in [N], s \in [T_1, t]$.

Then we have

$$
\frac{\exp(\langle \boldsymbol{q}_{\pm}^{(s)}, \boldsymbol{k}_{n,j}^{(s)} \rangle)}{\exp(\langle \boldsymbol{q}_{\pm}^{(s)}, \boldsymbol{k}_{\pm}^{(s)} \rangle) + \sum\limits_{j'=2}^{M} \exp(\langle \boldsymbol{q}_{\pm}^{(s)}, \boldsymbol{k}_{n,j'}^{(s)} \rangle)}
$$

$$
\leq \frac{\exp(\langle \boldsymbol{q}_{\pm}^{(s)}, \boldsymbol{k}_{n,j}^{(s)} \rangle)}{C \exp(\langle \boldsymbol{q}_{\pm}^{(s)}, \boldsymbol{k}_{\pm}^{(s)} \rangle)}
$$

$$
= \frac{1}{C \exp(\Lambda_{n,\pm,j}^{(s)})} \tag{253}
$$

$$
\leq \frac{1}{C \exp(\Lambda_{n,\pm,j}^{(T_1)}) + \frac{\eta^2 C_8 C \|\boldsymbol{\mu}\|_2^4 \|\boldsymbol{w}_O\|_2^2 d_h^{\frac{1}{2}}}{N \left( \log(6N^2 M^2/\delta) \right)^2} \cdot (s - T_1)(s - T_1 - 1)}
$$

$$
\leq \frac{1}{C_{13} + \frac{\eta^2 C_{13} \|\boldsymbol{\mu}\|_2^4 \|\boldsymbol{w}_O\|_2^2 d_h^{\frac{1}{2}}}{N \left( \log(6N^2 M^2/\delta) \right)^2} \cdot (s - T_1)(s - T_1 - 1)}.
$$

For the first inequality, by $\langle \boldsymbol{q}^{(T_1)}, \boldsymbol{k}^{(T_1)} \rangle = o(1)$ and the monotonicity of $\langle \boldsymbol{q}^{(s)}, \boldsymbol{k}^{(s)} \rangle$ ($\langle \boldsymbol{q}_{\pm}^{(s)}, \boldsymbol{k}_{\pm}^{(s)} \rangle$ is increasing and $\langle \boldsymbol{q}_{\pm}^{(s)}, \boldsymbol{k}_{n,j}^{(s)} \rangle$ is decreasing), there exist a constant C such that $C \exp(\langle \boldsymbol{q}_{\pm}^{(s)}, \boldsymbol{k}_{\pm}^{(s)} \rangle) \geq \exp(\langle \boldsymbol{q}_{\pm}^{(s)}, \boldsymbol{k}_{\pm}^{(s)} \rangle) + \sum\limits_{j'=2}^{M} \exp(\langle \boldsymbol{q}_{\pm}^{(s)}, \boldsymbol{k}_{n,j'}^{(s)} \rangle)$. The second inequality is by plugging (251). For the last inequality, by $\Lambda_{n,\pm,j}^{(T_1)} = o(1)$, there exist a constant $C_{13}$ such that $C_{13} \leq C \exp(\Lambda_{n,\pm,j}^{(T_1)})$ and $C_{13} \leq C_8 C$. Similarly, we have

$$
\frac{\exp(\langle \boldsymbol{q}_{n,i}^{(s)}, \boldsymbol{k}_{n,j}^{(s)} \rangle)}{\exp(\langle \boldsymbol{q}_{n,i}^{(s)}, \boldsymbol{k}_{+}^{(s)} \rangle) + \sum\limits_{j'=2}^{M} \exp(\langle \boldsymbol{q}_{n,i}^{(s)}, \boldsymbol{k}_{n,j'}^{(s)} \rangle)}
$$

$$
\leq \frac{1}{C \exp(\Lambda_{n,i,+,j}^{(s)})} \tag{254}
$$

$$
\leq \frac{1}{C_{13} + \frac{\eta^2 C_{13} \sigma_p^2 d \|\boldsymbol{\mu}\|_2^2 \|\boldsymbol{w}_O\|_2^2 d_h^{\frac{1}{2}}}{N \left( \log(6N^2 M^2/\delta) \right)^2} \cdot (s - T_1)(s - T_1 - 1)}.
$$

Plugging (250),(253) and (254) into the expressions of $\alpha$, $\beta$ we have

$$|\alpha_{+,+}^{(s)}| = \Big| \frac{\eta}{NM} \sum_{n \in S_+} -\ell_n'^{(s)} \|\boldsymbol{\mu}\|_2^2$$

$$\cdot \Big( V_+^{(s)} \big( \frac{\exp(\langle \boldsymbol{q}_+^{(s)}, \boldsymbol{k}_+^{(s)} \rangle)}{\exp(\langle \boldsymbol{q}_+^{(s)}, \boldsymbol{k}_+^{(s)} \rangle) + \sum\limits_{j=2}^{M} \exp(\langle \boldsymbol{q}_+^{(s)}, \boldsymbol{k}_{n,j}^{(s)} \rangle)}$$

$$- \big( \frac{\exp(\langle \boldsymbol{q}_+^{(s)}, \boldsymbol{k}_+^{(s)} \rangle)}{\exp(\langle \boldsymbol{q}_+^{(s)}, \boldsymbol{k}_+^{(s)} \rangle) + \sum\limits_{j=2}^{M} \exp(\langle \boldsymbol{q}_+^{(s)}, \boldsymbol{k}_{n,j}^{(s)} \rangle)} \big)^2 \big)$$

$$- \sum_{i=2}^{M} \big( V_{n,i}^{(s)} \cdot \frac{\exp(\langle \boldsymbol{q}_+^{(s)}, \boldsymbol{k}_+^{(s)} \rangle)}{\exp(\langle \boldsymbol{q}_+^{(s)}, \boldsymbol{k}_+^{(s)} \rangle) + \sum\limits_{j=2}^{M} \exp(\langle \boldsymbol{q}_+^{(s)}, \boldsymbol{k}_{n,j}^{(s)} \rangle)}$$

$$\cdot \frac{\exp(\langle \boldsymbol{q}_+^{(s)}, \boldsymbol{k}_{n,i}^{(s)} \rangle)}{\exp(\langle \boldsymbol{q}_+^{(s)}, \boldsymbol{k}_+^{(s)} \rangle) + \sum\limits_{j=2}^{M} \exp(\langle \boldsymbol{q}_+^{(s)}, \boldsymbol{k}_{n,j}^{(s)} \rangle)} \big) \Big) \Big| \qquad (255)$$

$$\leq \frac{\eta \|\boldsymbol{\mu}\|_2^2}{NM} \cdot \frac{3N}{4} \cdot \Big( O(d_h^{-\frac{1}{4}}) + \eta C_4 \|\boldsymbol{\mu}\|_2^2 \|\boldsymbol{w}_O\|_2^2 (s - T_1) \Big)$$

$$\cdot O\Big( \frac{1}{C_{13} + \frac{\eta^2 C_{13} \|\boldsymbol{\mu}\|_2^4 \|\boldsymbol{w}_O\|_2^2 d_h^{\frac{1}{2}}}{N \big( \log(6N^2 M^2/\delta) \big)^2} \cdot (s - T_1)(s - T_1 - 1)} \Big)$$

$$= O\Big( \frac{\eta \|\boldsymbol{\mu}\|_2^2 d_h^{-\frac{1}{4}}}{C_{13} + \frac{\eta^2 C_{13} \|\boldsymbol{\mu}\|_2^4 \|\boldsymbol{w}_O\|_2^2 d_h^{\frac{1}{2}}}{N \big( \log(6N^2 M^2/\delta) \big)^2} \cdot (s - T_1)(s - T_1 - 1)} \Big)$$

$$+ O\Big( \frac{\eta^2 \|\boldsymbol{\mu}\|_2^4 \|\boldsymbol{w}_O\|_2^2 (s - T_1)}{C_{13} + \frac{\eta^2 C_{13} \|\boldsymbol{\mu}\|_2^4 \|\boldsymbol{w}_O\|_2^2 d_h^{\frac{1}{2}}}{N \big( \log(6N^2 M^2/\delta) \big)^2} \cdot (s - T_1)(s - T_1 - 1)} \Big)$$

$$= O\Big( \eta \|\boldsymbol{\mu}\|_2^2 d_h^{-\frac{1}{4}} \Big) + O\Big( \frac{\eta^2 \|\boldsymbol{\mu}\|_2^4 \|\boldsymbol{w}_O\|_2^2 (s - T_1)}{C_{13} + \frac{\eta^2 C_{13} \|\boldsymbol{\mu}\|_2^4 \|\boldsymbol{w}_O\|_2^2 d_h^{\frac{1}{2}}}{N \big( \log(6N^2 M^2/\delta) \big)^2} \cdot (s - T_1)(s - T_1 - 1)} \Big).$$

where the third equality is by $\frac{\eta^2 C_{13}\|\boldsymbol{\mu}\|_2^4\|\boldsymbol{w}_O\|_2^2 d_h^{\frac{1}{2}}}{N\left(\log(6N^2M^2/\delta)\right)^2} \cdot (s-T_1)(s-T_1-1) \geq 0$ for $s \in [T_1, t]$. Next,

we give an upper bound for $\dfrac{\eta^2\|\boldsymbol{\mu}\|_2^4\|\boldsymbol{w}_O\|_2^2(s-T_1)}{C_{13}+\frac{\eta^2 C_{13}\|\boldsymbol{\mu}\|_2^4\|\boldsymbol{w}_O\|_2^2 d_h^{\frac{1}{2}}}{N\left(\log(6N^2M^2/\delta)\right)^2}\cdot(s-T_1)(s-T_1-1)}$ as follows:

$$
\begin{aligned}
&\frac{\eta^2\|\boldsymbol{\mu}\|_2^4\|\boldsymbol{w}_O\|_2^2(s-T_1)}{C_{13}+\frac{\eta^2 C_{13}\|\boldsymbol{\mu}\|_2^4\|\boldsymbol{w}_O\|_2^2 d_h^{\frac{1}{2}}}{N\left(\log(6N^2M^2/\delta)\right)^2}\cdot(s-T_1)(s-T_1-1)}\\
&=\frac{\eta^2\|\boldsymbol{\mu}\|_2^4\|\boldsymbol{w}_O\|_2^2}{\frac{C_{13}}{(s-T_1)}+\frac{\eta^2 C_{13}\|\boldsymbol{\mu}\|_2^4\|\boldsymbol{w}_O\|_2^2 d_h^{\frac{1}{2}}}{N\left(\log(6N^2M^2/\delta)\right)^2}\cdot(s-T_1)-\frac{\eta^2 C_{13}\|\boldsymbol{\mu}\|_2^4\|\boldsymbol{w}_O\|_2^2 d_h^{\frac{1}{2}}}{N\left(\log(6N^2M^2/\delta)\right)^2}}\\
&\leq\frac{\eta^2\|\boldsymbol{\mu}\|_2^4\|\boldsymbol{w}_O\|_2^2}{2\sqrt{\frac{\eta^2 C_{13}^2\|\boldsymbol{\mu}\|_2^4\|\boldsymbol{w}_O\|_2^2 d_h^{\frac{1}{2}}}{N\left(\log(6N^2M^2/\delta)\right)^2}}-\frac{\eta^2 C_{13}\|\boldsymbol{\mu}\|_2^4\|\boldsymbol{w}_O\|_2^2 d_h^{\frac{1}{2}}}{N\left(\log(6N^2M^2/\delta)\right)^2}}\\
&=\frac{\eta^2\|\boldsymbol{\mu}\|_2^4\|\boldsymbol{w}_O\|_2^2}{\frac{2\eta C_{13}\|\boldsymbol{\mu}\|_2^2\|\boldsymbol{w}_O\|_2 d_h^{\frac{1}{4}}}{N^{\frac{1}{2}}\left(\log(6N^2M^2/\delta)\right)}-\frac{\eta^2 C_{13}\|\boldsymbol{\mu}\|_2^4\|\boldsymbol{w}_O\|_2^2 d_h^{\frac{1}{2}}}{N\left(\log(6N^2M^2/\delta)\right)^2}}\\
&=\frac{\eta^2\|\boldsymbol{\mu}\|_2^4\|\boldsymbol{w}_O\|_2^2}{\Theta\left(\frac{\eta\|\boldsymbol{\mu}\|_2^2\|\boldsymbol{w}_O\|_2 d_h^{\frac{1}{4}}}{N^{\frac{1}{2}}\log(6N^2M^2/\delta)}\right)}\\
&=O\left(\eta\|\boldsymbol{\mu}\|_2^2 N^{\frac{1}{2}}d_h^{-\frac{1}{4}}\log(6N^2M^2/\delta)\right),
\end{aligned}
\tag{256}
$$

where the inequality is by $ax+\frac{b}{x}\geq 2\sqrt{ab}$ for $x>0$, the third equality is by absorbing the lower

order term $\frac{\eta^2 C_{13}\|\boldsymbol{\mu}\|_2^4\|\boldsymbol{w}_O\|_2^2 d_h^{\frac{1}{2}}}{N\left(\log(6N^2M^2/\delta)\right)^2}$, the last equality is by $\|\boldsymbol{w}_O\|_2=\Theta(1)$. Plugging this into (255) and get

$$
\begin{aligned}
|\alpha_{+,+}^{(s)}|&=O\left(\eta\|\boldsymbol{\mu}\|_2^2 d_h^{-\frac{1}{4}}\right)+O\left(\eta\|\boldsymbol{\mu}\|_2^2 N^{\frac{1}{2}}d_h^{-\frac{1}{4}}\log(6N^2M^2/\delta)\right)\\
&=O\left(\eta\|\boldsymbol{\mu}\|_2^2 N^{\frac{1}{2}}d_h^{-\frac{1}{4}}\log(6N^2M^2/\delta)\right).
\end{aligned}
\tag{257}
$$

Similarly, we have

$$
|\alpha_{-,-}^{(s)}|,|\beta_{+,+}^{(s)}|,|\beta_{-,-}^{(s)}|=O\left(\eta\|\boldsymbol{\mu}\|_2^2 N^{\frac{1}{2}}d_h^{-\frac{1}{4}}\log(6N^2M^2/\delta)\right),
\tag{258}
$$

$$
|\alpha_{n,+,i}^{(s)}|,|\alpha_{n,-,i}^{(s)}|=O\left(\eta\|\boldsymbol{\mu}\|_2^2 N^{-\frac{1}{2}}d_h^{-\frac{1}{4}}\log(6N^2M^2/\delta)\right),
\tag{259}
$$

$$
|\beta_{n,+,i}^{(s)}|,|\beta_{n,-,i}^{(s)}|=O\left(\frac{\eta\|\boldsymbol{\mu}\|_2^3\log(6N^2M^2/\delta)}{\sigma_p d^{\frac{1}{2}}N^{\frac{1}{2}}d_h^{\frac{1}{4}}}\right)=O\left(\eta\|\boldsymbol{\mu}\|_2^2\cdot\mathrm{SNR}\cdot N^{-\frac{1}{2}}d_h^{-\frac{1}{4}}\log(6N^2M^2/\delta)\right),
\tag{260}
$$

for $i\in[M]\backslash\{1\},n\in S_\pm$.

$$
|\alpha_{n,i,+}^{(s)}|,|\alpha_{n,i,-}^{(s)}|=O\left(\frac{\eta\|\boldsymbol{\mu}\|_2\sigma_p d^{\frac{1}{2}}\log(6N^2M^2/\delta)}{N^{\frac{1}{2}}d_h^{\frac{1}{4}}}\right)=O\left(\eta\|\boldsymbol{\mu}\|_2^2 d_h^{-\frac{1}{4}}\log(6N^2M^2/\delta)\right),
\tag{261}
$$

for $i\in[M]\backslash\{1\},n\in S_\pm$, the last equality is by $N\cdot\mathrm{SNR}^2\geq\Omega(1)$.

$$
|\beta_{n,i,+}^{(s)}|,|\beta_{n,i,-}^{(s)}|=O\left(\frac{\eta\sigma_p^2 d\log(6N^2M^2/\delta)}{N^{\frac{1}{2}}d_h^{\frac{1}{4}}}\right)=O\left(\eta\|\boldsymbol{\mu}\|_2^2 N^{\frac{1}{2}}d_h^{-\frac{1}{4}}\log(6N^2M^2/\delta)\right),
\tag{262}
$$

for $i \in [M]\backslash\{1\}, n \in S_\pm$, the last equality is by $N \cdot \mathrm{SNR}^2 \geq \Omega(1)$.

$$|\alpha_{n,i,n,j}^{(s)}|, |\beta_{n,j,n,i}^{(s)}| = O\Big(\frac{\eta\|\boldsymbol{\mu}\|_2\sigma_p d^{\frac{1}{2}} \log(6N^2M^2/\delta)}{N^{\frac{1}{2}}d_h^{\frac{1}{4}}}\Big) = O\Big(\eta\|\boldsymbol{\mu}\|_2^2 d_h^{-\frac{1}{4}}\log(6N^2M^2/\delta)\Big),$$

(263)

for $i, j \in [M]\backslash\{1\}, n \in [N]$, the last equality is by $N \cdot \mathrm{SNR}^2 \geq \Omega(1)$.

$$|\alpha_{n,i,n',j}^{(s)}|, |\beta_{n,j,n',i}^{(s)}| = O\Big(\frac{\eta\|\boldsymbol{\mu}\|_2\sigma_p \log(6N^2M^2/\delta)\log(4N^2M^2/\delta)}{N^{\frac{1}{2}}d_h^{\frac{1}{4}}}\Big) = O\Big(\eta\|\boldsymbol{\mu}\|_2^2 d^{-\frac{1}{2}}d_h^{-\frac{1}{4}}\big(\log(6N^2M^2/\delta)\big)^2\Big),$$

(264)

for $i, j \in [M]\backslash\{1\}, n, n' \in [N], n \neq n'$, the last equality is by $N \cdot \mathrm{SNR}^2 \geq \Omega(1)$. Taking a summation we obtain that

$$\sum_{s=T_1}^t |\alpha_{+,+}^{(s)}| = O\Big(\frac{1}{\eta\|\boldsymbol{\mu}\|_2^2\|\boldsymbol{w}_O\|_2^2 \log(6N^2M^2/\delta)}\Big) \cdot O\Big(\eta\|\boldsymbol{\mu}\|_2^2 N^{\frac{1}{2}}d_h^{-\frac{1}{4}}\log(6N^2M^2/\delta)\Big)$$
$$= O\Big(N^{\frac{1}{2}}d_h^{-\frac{1}{4}}\Big),$$

(265)

where the last equality is by $\|\boldsymbol{w}_O\| = \Theta(1)$. Similarly, we have

$$\sum_{s=T_1}^t |\alpha_{-,-}^{(s)}|, \sum_{s=T_1}^t |\beta_{+,+}^{(s)}|, \sum_{s=T_1}^t |\beta_{-,-}^{(s)}|, \sum_{s=T_1}^t |\beta_{n,i,+}^{(s)}|, \sum_{s=T_1}^t |\beta_{n,i,-}^{(s)}| = O\Big(N^{\frac{1}{2}}d_h^{-\frac{1}{4}}\Big),$$

(266)

for $i \in [M]\backslash\{1\}, n \in S_\pm$.

$$\sum_{s=T_1}^t |\alpha_{n,+,i}^{(s)}|, \sum_{s=T_1}^t |\alpha_{n,-,i}^{(s)}| = O\Big(N^{-\frac{1}{2}}d_h^{-\frac{1}{4}}\Big),$$

(267)

for $i \in [M]\backslash\{1\}, n \in S_\pm$.

$$\sum_{s=T_1}^t |\beta_{n,+,i}^{(s)}|, \sum_{s=T_1}^t |\beta_{n,-,i}^{(s)}| = O\Big(\mathrm{SNR} \cdot N^{-\frac{1}{2}}d_h^{-\frac{1}{4}}\Big)$$

(268)

for $i \in [M]\backslash\{1\}, n \in S_\pm$.

$$\sum_{s=T_1}^t |\alpha_{n,i,+}^{(s)}|, \sum_{s=T_1}^t |\alpha_{n,i,-}^{(s)}|, \sum_{s=T_1}^t |\alpha_{n,i,n,j}^{(s)}|, \sum_{s=T_1}^t |\beta_{n,j,n,i}^{(s)}| = O\Big(d_h^{-\frac{1}{4}}\Big)$$

(269)

for $i, j \in [M]\backslash\{1\}, n \in S_\pm$.

$$\sum_{s=T_1}^t |\alpha_{n,i,n',j}^{(s)}|, \sum_{s=T_1}^t |\beta_{n,j,n',i}^{(s)}| = O\Big(d^{-\frac{1}{2}}d_h^{-\frac{1}{4}}\log(6N^2M^2/\delta)\Big)$$

(270)

for $i, j \in [M]\backslash\{1\}, n, n' \in [N], n \neq n'$.

With these sums of $\alpha$ and $\beta$ above, we can easily prove **Claim** 3 and **Claim** 4.

### F.8 Proof of Claim 3

In this subsection, we assume that $\mathcal{E}(T_1), \ldots, \mathcal{E}(t)$ hold, and then proof that $\mathcal{C}(t+1)$ is true with the result of F.7.

$$\Big|\|\boldsymbol{q}_+^{(t+1)}\|_2^2 - \|\boldsymbol{q}_+^{(T_1)}\|_2^2\Big| \leq \sum_{s=T_1}^t \Big|\|\boldsymbol{q}_+^{(s+1)}\|_2^2 - \|\boldsymbol{q}_+^{(s)}\|_2^2\Big|$$

$$\leq \sum_{s=T_1}^{t} \left| 2\alpha_{+,+}^{(s)} \langle \boldsymbol{q}_{+}^{(s)}, \boldsymbol{k}_{+}^{(s)} \rangle + 2 \sum_{n \in S_+} \sum_{i=2}^{M} \alpha_{n,+,i}^{(s)} \langle \boldsymbol{q}_{+}^{(s)}, \boldsymbol{k}_{n,i}^{(s)} \rangle \right.$$

$$+ \left( \alpha_{+,+}^{(s)} \boldsymbol{k}_{+}^{(s)} + \sum_{n \in S_+} \sum_{i=2}^{M} \alpha_{n,+,i}^{(s)} \boldsymbol{k}_{n,i}^{(s)} \right)$$

$$\left. \cdot \left( \alpha_{+,+}^{(s)} \boldsymbol{k}_{+}^{(s)\top} + \sum_{n \in S_+} \sum_{i=2}^{M} \alpha_{n,+,i}^{(s)} \boldsymbol{k}_{n,i}^{(s)\top} \right) \right|$$

$$\leq 2 \sum_{s=T_1}^{t} |\alpha_{+,+}^{(s)}| |\langle \boldsymbol{q}_{+}^{(s)}, \boldsymbol{k}_{+}^{(s)} \rangle| + 2 \sum_{n \in S_+} \sum_{i=2}^{M} \sum_{s=T_1}^{t} |\alpha_{n,+,i}^{(s)}| |\langle \boldsymbol{q}_{+}^{(s)}, \boldsymbol{k}_{n,i}^{(s)} \rangle|$$

$$+ \{lower\ order\ term\}$$

$$= O\left( N^{\frac{1}{2}} d_h^{-\frac{1}{4}} \right) \cdot O(\log(d_h^{\frac{1}{2}})) + N \cdot M \cdot O\left( N^{-\frac{1}{2}} d_h^{-\frac{1}{4}} \right) \cdot O(\log(d_h^{\frac{1}{2}}))$$

$$= O\left( N^{\frac{1}{2}} d_h^{-\frac{1}{4}} \log(d_h^{\frac{1}{2}}) \right)$$

where the first inequality is by triangle inequality, the second inequality is by 199. Since $\sigma_h^2 \geq \left( \max\{\sigma_p^2 d, \|\boldsymbol{\mu}\|_2^2\} \right)^{-1} \cdot d_h^{-\frac{1}{2}} (\log(6N^2 M^2/\delta))^{-2}$ and $d_h = \widetilde{\Omega}\left( \max\{\text{SNR}^4, \text{SNR}^{-4}\} N^2 \epsilon^{-2} \right)$, we have $N^{\frac{1}{2}} d_h^{-\frac{1}{4}} \log(d_h^{\frac{1}{2}}) = o(\|\boldsymbol{\mu}\|_2^2 \sigma_h^2 d_h)$, so $\|\boldsymbol{q}_{+}^{(t+1)}\|_2^2 = \|\boldsymbol{q}_{+}^{(T_1)}\|_2^2 + o(\|\boldsymbol{\mu}\|_2^2 \sigma_h^2 d_h) = \Theta(\|\boldsymbol{\mu}\|_2^2 \sigma_h^2 d_h)$. Similarly, we have

$$\left| \|\boldsymbol{q}_{-}^{(t+1)}\|_2^2 - \|\boldsymbol{q}_{-}^{(T_1)}\|_2^2 \right| = O\left( N^{\frac{1}{2}} d_h^{-\frac{1}{4}} \log(d_h^{\frac{1}{2}}) \right) = o(\|\boldsymbol{\mu}\|_2^2 \sigma_h^2 d_h),$$

$$\left| \|\boldsymbol{k}_{\pm}^{(t+1)}\|_2^2 - \|\boldsymbol{k}_{\pm}^{(T_1)}\|_2^2 \right| = O\left( (1 + \text{SNR}) N^{\frac{1}{2}} d_h^{-\frac{1}{4}} \log(d_h^{\frac{1}{2}}) \right) = o(\|\boldsymbol{\mu}\|_2^2 \sigma_h^2 d_h),$$

$$\left| \|\boldsymbol{q}_{n,i}^{(t+1)}\|_2^2 - \|\boldsymbol{q}_{n,i}^{(T_1)}\|_2^2 \right| = O\left( d_h^{-\frac{1}{4}} \log(d_h^{\frac{1}{2}}) \right) = o(\sigma_p^2 \sigma_h^2 d d_h),$$

$$\left| \|\boldsymbol{k}_{n,i}^{(t+1)}\|_2^2 - \|\boldsymbol{k}_{n,i}^{(T_1)}\|_2^2 \right| = O\left( N^{\frac{1}{2}} d_h^{-\frac{1}{4}} \log(d_h^{\frac{1}{2}}) \right) = o(\sigma_p^2 \sigma_h^2 d d_h),$$

so we have

$$\|\boldsymbol{q}_{\pm}^{(t+1)}\|_2^2, \|\boldsymbol{k}_{\pm}^{(t+1)}\|_2^2 = \Theta(\|\boldsymbol{\mu}\|_2^2 \sigma_h^2 d_h),$$

$$\|\boldsymbol{q}_{n,i}^{(t+1)}\|_2^2, \|\boldsymbol{k}_{n,i}^{(t+1)}\|_2^2 = \Theta(\sigma_p^2 \sigma_h^2 d d_h)$$

for $i \in [M] \backslash \{1\}, n \in [N]$.

$$|\langle \boldsymbol{q}_{+}^{(t+1)}, \boldsymbol{q}_{-}^{(t+1)} \rangle| \leq |\langle \boldsymbol{q}_{+}^{(T_1)}, \boldsymbol{q}_{-}^{(T_1)} \rangle| + \sum_{s=T_1}^{t} \left| \langle \boldsymbol{q}_{+}^{(s+1)}, \boldsymbol{q}_{-}^{(s+1)} \rangle - \langle \boldsymbol{q}_{+}^{(s)}, \boldsymbol{q}_{-}^{(s)} \rangle \right|$$

$$\leq |\langle \boldsymbol{q}_{+}^{(T_1)}, \boldsymbol{q}_{-}^{(T_1)} \rangle|$$

$$+ \sum_{s=T_1}^{t} \left| \alpha_{+,+}^{(s)} \langle \boldsymbol{q}_{-}^{(s)}, \boldsymbol{k}_{+}^{(s)} \rangle + \sum_{n \in S_+} \sum_{i=2}^{M} \alpha_{n,+,i}^{(s)} \langle \boldsymbol{q}_{-}^{(s)}, \boldsymbol{k}_{n,i}^{(s)} \rangle \right.$$

$$+ \alpha_{-,-}^{(s)} \langle \boldsymbol{q}_{+}^{(s)}, \boldsymbol{k}_{-}^{(s)} \rangle + \sum_{n \in S_-} \sum_{i=2}^{M} \alpha_{n,-,i}^{(s)} \langle \boldsymbol{q}_{+}^{(s)}, \boldsymbol{k}_{n,i}^{(s)} \rangle$$

$$+ \left( \alpha_{+,+}^{(s)} \boldsymbol{k}_{+}^{(s)} + \sum_{n \in S_+} \sum_{i=2}^{M} \alpha_{n,+,i}^{(s)} \boldsymbol{k}_{n,i}^{(s)} \right)$$

$$\left. \cdot \left( \alpha_{-,-}^{(s)} \boldsymbol{k}_{-}^{(s)\top} + \sum_{n \in S_-} \sum_{i=2}^{M} \alpha_{n,-,i}^{(s)} \boldsymbol{k}_{n,i}^{(s)\top} \right) \right|$$

$$\leq |\langle \boldsymbol{q}_+^{(T_1)}, \boldsymbol{q}_-^{(T_1)}\rangle|$$

$$+ \sum_{s=T_1}^{t} |\alpha_{+,+}^{(s)}||\langle \boldsymbol{q}_-^{(s)}, \boldsymbol{k}_+^{(s)}\rangle| + \sum_{n\in S_+}^{M}\sum_{i=2}^{t}\sum_{s=T_1} |\alpha_{n,+,i}^{(s)}||\langle \boldsymbol{q}_-^{(s)}, \boldsymbol{k}_{n,i}^{(s)}\rangle|$$

$$+ \sum_{s=T_1}^{t} |\alpha_{-,-}^{(s)}||\langle \boldsymbol{q}_+^{(s)}, \boldsymbol{k}_-^{(s)}\rangle| + \sum_{n\in S_-}^{M}\sum_{i=2}^{t}\sum_{s=T_1} |\alpha_{n,-,i}^{(s)}||\langle \boldsymbol{q}_+^{(s)}, \boldsymbol{k}_{n,i}^{(s)}\rangle|$$

$$+ \{lower\ order\ term\}$$

$$\leq |\langle \boldsymbol{q}_+^{(T_1)}, \boldsymbol{q}_-^{(T_1)}\rangle|$$

$$+ O\left(N^{\frac{1}{2}}d_h^{-\frac{1}{4}}\right)\cdot o(1) + N\cdot M\cdot O\left(N^{-\frac{1}{2}}d_h^{-\frac{1}{4}}\right)\cdot \log(d_h^{\frac{1}{2}})$$

$$= |\langle \boldsymbol{q}_+^{(T_1)}, \boldsymbol{q}_-^{(T_1)}\rangle| + O\left(N^{\frac{1}{2}}d_h^{-\frac{1}{4}}\log(d_h^{\frac{1}{2}})\right)$$

$$= o(1),$$

where the first inequality is triangle inequality, the second inequality is by (198), the last equality is by $d_h = \widetilde{\Omega}\left(\max\{\mathrm{SNR}^4, \mathrm{SNR}^{-4}\}N^2\epsilon^{-2}\right)$.

$$|\langle \boldsymbol{q}_+^{(t+1)}, \boldsymbol{q}_{n,i}^{(t+1)}\rangle| \leq |\langle \boldsymbol{q}_+^{(T_1)}, \boldsymbol{q}_{n,i}^{(T_1)}\rangle| + \sum_{s=T_1}^{t}\left|\langle \boldsymbol{q}_+^{(s+1)}, \boldsymbol{q}_{n,i}^{(s+1)}\rangle - \langle \boldsymbol{q}_+^{(s)}, \boldsymbol{q}_{n,i}^{(s)}\rangle\right|$$

$$\leq |\langle \boldsymbol{q}_+^{(T_1)}, \boldsymbol{q}_{n,i}^{(T_1)}\rangle|$$

$$+ \sum_{s=T_1}^{t}\left|\alpha_{+,+}^{(s)}\langle \boldsymbol{q}_{n,i}^{(s)}, \boldsymbol{k}_+^{(s)}\rangle + \sum_{n'\in S_+}\sum_{l=2}^{M}\alpha_{n',+,l}^{(s)}\langle \boldsymbol{q}_{n,i}^{(s)}, \boldsymbol{k}_{n',l}^{(s)}\rangle\right.$$

$$+ \alpha_{n,i,+}^{(s)}\langle \boldsymbol{q}_+^{(s)}, \boldsymbol{k}_+^{(s)}\rangle + \alpha_{n,i,-}^{(s)}\langle \boldsymbol{q}_+^{(s)}, \boldsymbol{k}_-^{(s)}\rangle + \sum_{n'=1}^{N}\sum_{l=2}^{M}\alpha_{n,i,n',l}^{(s)}\langle \boldsymbol{q}_+^{(s)}, \boldsymbol{k}_{n',l}^{(s)}\rangle$$

$$+ \left(\alpha_{+,+}^{(s)}\boldsymbol{k}_+^{(s)} + \sum_{n\in S_+}\sum_{i=2}^{M}\alpha_{n,+,i}^{(s)}\boldsymbol{k}_{n,i}^{(s)}\right)$$

$$\cdot\left.\left(\alpha_{n,i,+}^{(s)}\boldsymbol{k}_+^{(s)\top} + \alpha_{n,i,-}^{(s)}\boldsymbol{k}_-^{(s)\top} + \sum_{n'=1}^{N}\sum_{l=2}^{M}\alpha_{n,i,n',l}^{(s)}\boldsymbol{k}_{n',l}^{(s)\top}\right)\right|$$

$$\leq |\langle \boldsymbol{q}_+^{(T_1)}, \boldsymbol{q}_{n,i}^{(T_1)}\rangle|$$

$$+ \sum_{s=T_1}^{t}|\alpha_{+,+}^{(s)}||\langle \boldsymbol{q}_{n,i}^{(s)}, \boldsymbol{k}_+^{(s)}\rangle| + \sum_{l=2}^{M}\sum_{s=T_1}^{t}|\alpha_{n,+,l}^{(s)}||\langle \boldsymbol{q}_{n,i}^{(s)}, \boldsymbol{k}_{n,l}^{(s)}\rangle| + \sum_{n'\in S_+\wedge n'\neq n}\sum_{l=2}^{M}\sum_{s=T_1}^{t}|\alpha_{n',+,l}^{(s)}||\langle \boldsymbol{q}_{n,i}^{(s)}, \boldsymbol{k}_{n',l}^{(s)}\rangle|$$

$$+ \sum_{s=T_1}^{t}|\alpha_{n,i,+}^{(s)}||\langle \boldsymbol{q}_+^{(s)}, \boldsymbol{k}_+^{(s)}\rangle| + \sum_{s=T_1}^{t}|\alpha_{n,i,-}^{(s)}||\langle \boldsymbol{q}_+^{(s)}, \boldsymbol{k}_-^{(s)}\rangle| + \sum_{l=2}^{M}\sum_{s=T_1}^{t}|\alpha_{n,i,n,l}^{(s)}||\langle \boldsymbol{q}_+^{(s)}, \boldsymbol{k}_{n,l}^{(s)}\rangle|$$

$$+ \sum_{n'\neq n}^{M}\sum_{l=2}^{t}\sum_{s=T_1}|\alpha_{n,i,n',l}^{(s)}||\langle \boldsymbol{q}_+^{(s)}, \boldsymbol{k}_{n',l}^{(s)}\rangle|$$

$$+ \{lower\ order\ term\}$$

$$\leq |\langle \boldsymbol{q}_+^{(T_1)}, \boldsymbol{q}_{n,i}^{(T_1)}\rangle|$$

$$+ O\left(N^{\frac{1}{2}}d_h^{-\frac{1}{4}}\right)\cdot \log(d_h^{\frac{1}{2}}) + M\cdot O\left(N^{-\frac{1}{2}}d_h^{-\frac{1}{4}}\right)\cdot \log(d_h^{\frac{1}{2}})$$

$$+ N\cdot M\cdot O\left(N^{-\frac{1}{2}}d_h^{-\frac{1}{4}}\right)\cdot o(1) + O\left(d_h^{-\frac{1}{4}}\right)\cdot \log(d_h^{\frac{1}{2}})$$

$$+ M\cdot O\left(d_h^{-\frac{1}{4}}\right)\cdot \log(d_h^{\frac{1}{2}}) + N\cdot M\cdot O\left(d^{-\frac{1}{2}}d_h^{-\frac{1}{4}}\log(6N^2M^2/\delta)\right)\cdot \log(d_h^{\frac{1}{2}})$$

$$= |\langle \boldsymbol{q}_+^{(T_1)}, \boldsymbol{q}_{n,i}^{(T_1)}\rangle| + O\left(N^{\frac{1}{2}}d_h^{-\frac{1}{4}}\right) + O\left(Nd^{-\frac{1}{2}}d_h^{-\frac{1}{4}}\log(6N^2M^2/\delta)\log(d_h^{\frac{1}{2}})\right)$$

$$= o(1),$$

where the first inequality is triangle inequality, the second inequality is by (202), the last equality is by $d_h = \widetilde{\Omega}\left(\max\{\text{SNR}^4, \text{SNR}^{-4}\}N^2\epsilon^{-2}\right)$ and $d = \widetilde{\Omega}\left(\epsilon^{-2}N^2d_h\right)$. Similarly, we have $|\langle \boldsymbol{q}_-^{(t+1)}, \boldsymbol{q}_{n,i}^{(t+1)}\rangle| = o(1)$.

$$|\langle \boldsymbol{q}_{n,i}^{(t+1)}, \boldsymbol{q}_{n,j}^{(t+1)}\rangle| \le |\langle \boldsymbol{q}_{n,i}^{(T_1)}, \boldsymbol{q}_{n,j}^{(T_1)}\rangle| + \sum_{s=T_1}^{t}\left|\langle \boldsymbol{q}_{n,i}^{(s+1)}, \boldsymbol{q}_{n,j}^{(s+1)}\rangle - \langle \boldsymbol{q}_{n,i}^{(s)}, \boldsymbol{q}_{n,j}^{(s)}\rangle\right|$$

$$\le |\langle \boldsymbol{q}_{n,i}^{(T_1)}, \boldsymbol{q}_{n,j}^{(T_1)}\rangle|$$

$$+ \sum_{s=T_1}^{t}\left|\alpha_{n,i,+}^{(s)}\langle \boldsymbol{q}_{n,j}^{(s)}, \boldsymbol{k}_+^{(s)}\rangle + \alpha_{n,i,-}^{(s)}\langle \boldsymbol{q}_{n,j}^{(s)}, \boldsymbol{k}_-^{(s)}\rangle + \sum_{n'=1}^{N}\sum_{l=2}^{M}\alpha_{n,i,n',l}^{(s)}\langle \boldsymbol{q}_{n,j}^{(s)}, \boldsymbol{k}_{n',l}^{(s)}\rangle\right.$$

$$+ \alpha_{n,j,+}^{(s)}\langle \boldsymbol{q}_{n,i}^{(s)}, \boldsymbol{k}_+^{(s)}\rangle + \alpha_{n,j,-}^{(s)}\langle \boldsymbol{q}_{n,i}^{(s)}, \boldsymbol{k}_-^{(s)}\rangle + \sum_{n'=1}^{N}\sum_{l=2}^{M}\alpha_{n,j,n',l}^{(s)}\langle \boldsymbol{q}_{n,i}^{(s)}, \boldsymbol{k}_{n',l}^{(s)}\rangle$$

$$+ \left(\alpha_{n,i,+}^{(s)}\boldsymbol{k}_+^{(s)} + \alpha_{n,i,-}^{(s)}\boldsymbol{k}_-^{(s)} + \sum_{n'=1}^{N}\sum_{l=2}^{M}\alpha_{n,i,n',l}^{(s)}\boldsymbol{k}_{n',l}^{(s)}\right)$$

$$\left.\cdot \left(\alpha_{n,j,+}^{(s)}\boldsymbol{k}_+^{(s)\top} + \alpha_{n,j,-}^{(s)}\boldsymbol{k}_-^{(s)\top} + \sum_{n'=1}^{N}\sum_{l=2}^{M}\alpha_{n,j,n',l}^{(s)}\boldsymbol{k}_{n',l}^{(s)\top}\right)\right|$$

$$\le |\langle \boldsymbol{q}_{n,i}^{(T_1)}, \boldsymbol{q}_{n,j}^{(T_1)}\rangle|$$

$$+ \sum_{s=T_1}^{t}|\alpha_{n,i,+}^{(s)}||\langle \boldsymbol{q}_{n,j}^{(s)}, \boldsymbol{k}_+^{(s)}\rangle| + \sum_{s=T_1}^{t}|\alpha_{n,i,-}^{(s)}||\langle \boldsymbol{q}_{n,j}^{(s)}, \boldsymbol{k}_-^{(s)}\rangle| + \sum_{l=2}^{M}\sum_{s=T_1}^{t}|\alpha_{n,i,n,l}^{(s)}||\langle \boldsymbol{q}_{n,j}^{(s)}, \boldsymbol{k}_{n,l}^{(s)}\rangle|$$

$$+ \sum_{n'\neq n}\sum_{l=2}^{M}\sum_{s=T_1}^{t}|\alpha_{n,i,n',l}^{(s)}||\langle \boldsymbol{q}_{n,j}^{(s)}, \boldsymbol{k}_{n',l}^{(s)}\rangle| + \sum_{s=T_1}^{t}|\alpha_{n,j,+}^{(s)}||\langle \boldsymbol{q}_{n,i}^{(s)}, \boldsymbol{k}_+^{(s)}\rangle| + \sum_{s=T_1}^{t}|\alpha_{n,j,-}^{(s)}||\langle \boldsymbol{q}_{n,i}^{(s)}, \boldsymbol{k}_-^{(s)}\rangle|$$

$$+ \sum_{l=2}^{M}\sum_{s=T_1}^{t}|\alpha_{n,j,n,l}^{(s)}||\langle \boldsymbol{q}_{n,i}^{(s)}, \boldsymbol{k}_{n,l}^{(s)}\rangle| + \sum_{n'\neq n}\sum_{l=2}^{M}\sum_{s=T_1}^{t}|\alpha_{n,j,n',l}^{(s)}||\langle \boldsymbol{q}_{n,i}^{(s)}, \boldsymbol{k}_{n',l}^{(s)}\rangle|$$

$$+ \{lower\ order\ term\}$$

$$\le |\langle \boldsymbol{q}_{n,i}^{(T_1)}, \boldsymbol{q}_{n,j}^{(T_1)}\rangle|$$

$$+ O\left(d_h^{-\frac{1}{4}}\right)\cdot \log(d_h^{\frac{1}{2}}) + M\cdot O\left(d_h^{-\frac{1}{4}}\right)\cdot \log(d_h^{\frac{1}{2}})$$

$$+ N\cdot M\cdot O\left(d^{-\frac{1}{2}}d_h^{-\frac{1}{4}}\log(6N^2M^2/\delta)\right)\cdot o(1)$$

$$= |\langle \boldsymbol{q}_{n,i}^{(T_1)}, \boldsymbol{q}_{n,j}^{(T_1)}\rangle| + O\left(d_h^{-\frac{1}{4}}\log(d_h^{\frac{1}{2}})\right) + o\left(Nd^{-\frac{1}{2}}d_h^{-\frac{1}{4}}\log(6N^2M^2/\delta)\right)$$

$$= o(1)$$

for $i, j \in [M]\backslash\{1\}, i \neq j, n \in [N]$. The first inequality is triangle inequality, the second inequality is by (204), the last equality is by $d_h = \widetilde{\Omega}\left(\max\{\text{SNR}^4, \text{SNR}^{-4}\}N^2\epsilon^{-2}\right)$ and $d = \widetilde{\Omega}\left(\epsilon^{-2}N^2d_h\right)$.

$$|\langle \boldsymbol{q}_{n,i}^{(t+1)}, \boldsymbol{q}_{\overline{n},j}^{(t+1)}\rangle| \le |\langle \boldsymbol{q}_{n,i}^{(T_1)}, \boldsymbol{q}_{\overline{n},j}^{(T_1)}\rangle| + \sum_{s=T_1}^{t}\left|\langle \boldsymbol{q}_{n,i}^{(s+1)}, \boldsymbol{q}_{\overline{n},j}^{(s+1)}\rangle - \langle \boldsymbol{q}_{n,i}^{(s)}, \boldsymbol{q}_{\overline{n},j}^{(s)}\rangle\right|$$

$$\le |\langle \boldsymbol{q}_{n,i}^{(T_1)}, \boldsymbol{q}_{\overline{n},j}^{(T_1)}\rangle|$$

$$+ \sum_{s=T_1}^{t}\left|\alpha_{n,i,+}^{(s)}\langle \boldsymbol{q}_{\overline{n},j}^{(s)}, \boldsymbol{k}_+^{(s)}\rangle + \alpha_{n,i,-}^{(s)}\langle \boldsymbol{q}_{\overline{n},j}^{(s)}, \boldsymbol{k}_-^{(s)}\rangle + \sum_{n'=1}^{N}\sum_{l=2}^{M}\alpha_{n,i,n',l}^{(s)}\langle \boldsymbol{q}_{\overline{n},j}^{(s)}, \boldsymbol{k}_{n',l}^{(s)}\rangle\right.$$

$$+ \alpha^{(s)}_{\overline{n},j,+}\langle \boldsymbol{q}^{(s)}_{n,i}, \boldsymbol{k}^{(s)}_+\rangle + \alpha^{(s)}_{\overline{n},j,-}\langle \boldsymbol{q}^{(s)}_{n,i}, \boldsymbol{k}^{(s)}_-\rangle + \sum_{n'=1}^{N}\sum_{l=2}^{M}\alpha^{(s)}_{\overline{n},j,n',l}\langle \boldsymbol{q}^{(s)}_{n,i}, \boldsymbol{k}^{(s)}_{n',l}\rangle$$

$$+ \left(\alpha^{(s)}_{n,i,+}\boldsymbol{k}^{(s)}_+ + \alpha^{(s)}_{n,i,-}\boldsymbol{k}^{(s)}_- + \sum_{n'=1}^{N}\sum_{l=2}^{M}\alpha^{(s)}_{n,i,n',l}\boldsymbol{k}^{(s)}_{n',l}\right)$$

$$\cdot \left(\alpha^{(s)}_{\overline{n},j,+}\boldsymbol{k}^{(s)\top}_+ + \alpha^{(s)}_{\overline{n},j,-}\boldsymbol{k}^{(s)\top}_- + \sum_{n'=1}^{N}\sum_{l=2}^{M}\alpha^{(s)}_{\overline{n},j,n',l}\boldsymbol{k}^{(s)\top}_{n',l}\right)\Bigg|$$

$$\leq |\langle \boldsymbol{q}^{(T_1)}_{n,i}, \boldsymbol{q}^{(T_1)}_{\overline{n},j}\rangle|$$

$$+ \sum_{s=T_1}^{t}|\alpha^{(s)}_{n,i,+}||\langle \boldsymbol{q}^{(s)}_{\overline{n},j}, \boldsymbol{k}^{(s)}_+\rangle| + \sum_{s=T_1}^{t}|\alpha^{(s)}_{n,i,-}||\langle \boldsymbol{q}^{(s)}_{\overline{n},j}, \boldsymbol{k}^{(s)}_-\rangle| + \sum_{l=2}^{M}\sum_{s=T_1}^{t}|\alpha^{(s)}_{n,i,\overline{n},l}||\langle \boldsymbol{q}^{(s)}_{\overline{n},j}, \boldsymbol{k}^{(s)}_{\overline{n},l}\rangle|$$

$$+ \sum_{l=2}^{M}\sum_{s=T_1}^{t}|\alpha^{(s)}_{n,i,n,l}||\langle \boldsymbol{q}^{(s)}_{\overline{n},j}, \boldsymbol{k}^{(s)}_{n,l}\rangle| + \sum_{n'\neq n \wedge n'\neq \overline{n}}\sum_{l=2}^{M}\sum_{s=T_1}^{t}|\alpha^{(s)}_{n,i,n',l}||\langle \boldsymbol{q}^{(s)}_{\overline{n},j}, \boldsymbol{k}^{(s)}_{n',l}\rangle|$$

$$+ \sum_{s=T_1}^{t}|\alpha^{(s)}_{\overline{n},j,+}||\langle \boldsymbol{q}^{(s)}_{n,i}, \boldsymbol{k}^{(s)}_+\rangle| + \sum_{s=T_1}^{t}|\alpha^{(s)}_{\overline{n},j,-}||\langle \boldsymbol{q}^{(s)}_{n,i}, \boldsymbol{k}^{(s)}_-\rangle| + \sum_{l=2}^{M}\sum_{s=T_1}^{t}|\alpha^{(s)}_{\overline{n},j,n,l}||\langle \boldsymbol{q}^{(s)}_{n,i}, \boldsymbol{k}^{(s)}_{n,l}\rangle|$$

$$+ \sum_{l=2}^{M}\sum_{s=T_1}^{t}|\alpha^{(s)}_{\overline{n},j,\overline{n},l}||\langle \boldsymbol{q}^{(s)}_{n,i}, \boldsymbol{k}^{(s)}_{\overline{n},l}\rangle| + \sum_{n'\neq n \wedge n'\neq \overline{n}}\sum_{l=2}^{M}\sum_{s=T_1}^{t}|\alpha^{(s)}_{\overline{n},j,n',l}||\langle \boldsymbol{q}^{(s)}_{n,i}, \boldsymbol{k}^{(s)}_{n',l}\rangle|$$

$$+ \{lower\ order\ term\}$$

$$= |\langle \boldsymbol{q}^{(T_1)}_{n,i}, \boldsymbol{q}^{(T_1)}_{\overline{n},j}\rangle|$$

$$+ O\left(d_h^{-\frac{1}{4}}\right)\cdot\log(d_h^{\frac{1}{2}}) + M\cdot O\left(d^{-\frac{1}{2}}d_h^{-\frac{1}{4}}\log(6N^2M^2/\delta)\right)\cdot\log(d_h^{\frac{1}{2}})$$

$$+ M\cdot O\left(d_h^{-\frac{1}{4}}\right)\cdot o(1) + N\cdot M\cdot O\left(d^{-\frac{1}{2}}d_h^{-\frac{1}{4}}\log(6N^2M^2/\delta)\right)\cdot o(1)$$

$$= |\langle \boldsymbol{q}^{(T_1)}_{n,i}, \boldsymbol{q}^{(T_1)}_{\overline{n},j}\rangle| + O\left(d_h^{-\frac{1}{4}}\log(d_h^{\frac{1}{2}})\right) + o\left(Nd^{-\frac{1}{2}}d_h^{-\frac{1}{4}}\log(6N^2M^2/\delta)\right)$$

$$= o(1)$$

for $i,j \in [M]\backslash\{1\}, n,\overline{n} \in [N], n \neq \overline{n}$. The first inequality is triangle inequality, the second inequality is by (205), the last equality is by $d_h = \widetilde{\Omega}\left(\max\{\text{SNR}^4, \text{SNR}^{-4}\}N^2\epsilon^{-2}\right)$ and $d = \widetilde{\Omega}\left(\epsilon^{-2}N^2d_h\right)$.

$$|\langle \boldsymbol{k}^{(t+1)}_+, \boldsymbol{k}^{(t+1)}_-\rangle| \leq |\langle \boldsymbol{k}^{(T_1)}_+, \boldsymbol{k}^{(T_1)}_-\rangle| + \sum_{s=T_1}^{t}\left|\langle \boldsymbol{k}^{(s+1)}_+, \boldsymbol{k}^{(s+1)}_-\rangle - \langle \boldsymbol{k}^{(s)}_+, \boldsymbol{k}^{(s)}_-\rangle\right|$$

$$\leq |\langle \boldsymbol{k}^{(T_1)}_+, \boldsymbol{k}^{(T_1)}_-\rangle|$$

$$+ \sum_{s=T_1}^{t}\left|\beta^{(s)}_{+,+}\langle \boldsymbol{q}^{(s)}_+, \boldsymbol{k}^{(s)}_-\rangle + \sum_{n\in S_+}\sum_{i=2}^{M}\beta^{(s)}_{n,+,i}\langle \boldsymbol{q}^{(s)}_{n,i}, \boldsymbol{k}^{(s)}_-\rangle\right.$$

$$+ \beta^{(s)}_{-,-}\langle \boldsymbol{q}^{(s)}_-, \boldsymbol{k}^{(s)}_+\rangle + \sum_{n\in S_-}\sum_{i=2}^{M}\beta^{(s)}_{n,-,i}\langle \boldsymbol{q}^{(s)}_{n,i}, \boldsymbol{k}^{(s)}_+\rangle$$

$$+ \left(\beta^{(s)}_{+,+}\boldsymbol{q}^{(s)}_+ + \sum_{n\in S_+}\sum_{i=2}^{M}\beta^{(s)}_{n,+,i}\boldsymbol{q}^{(s)}_{n,i}\right)$$

$$\left.\cdot \left(\beta^{(s)}_{-,-}\boldsymbol{q}^{(s)\top}_- + \sum_{n\in S_-}\sum_{i=2}^{M}\beta^{(s)}_{n,-,i}\boldsymbol{q}^{(s)\top}_{n,i}\right)\right|$$

$$\leq |\langle \boldsymbol{k}^{(T_1)}_+, \boldsymbol{k}^{(T_1)}_-\rangle|$$

$$+ \sum_{s=T_1}^{t} |\beta_{+,+}^{(s)}||\langle \boldsymbol{q}_+^{(s)}, \boldsymbol{k}_-^{(s)}\rangle| + \sum_{n\in S_+} \sum_{i=2}^{M} \sum_{s=T_1}^{t} |\beta_{n,+,i}^{(s)}||\langle \boldsymbol{q}_{n,i}^{(s)}, \boldsymbol{k}_-^{(s)}\rangle|$$

$$+ \sum_{s=T_1}^{t} |\beta_{-,-}^{(s)}||\langle \boldsymbol{q}_-^{(s)}, \boldsymbol{k}_+^{(s)}\rangle| + \sum_{n\in S_-} \sum_{i=2}^{M} \sum_{s=T_1}^{t} |\beta_{n,-,i}^{(s)}||\langle \boldsymbol{q}_{n,i}^{(s)}, \boldsymbol{k}_+^{(s)}\rangle|$$

$$+ \{lower\ order\ term\}$$

$$= |\langle \boldsymbol{k}_+^{(T_1)}, \boldsymbol{k}_-^{(T_1)}\rangle|$$

$$+ O\left(N^{\frac{1}{2}} d_h^{-\frac{1}{4}}\right) \cdot \log(d_h^{\frac{1}{2}}) + N \cdot M \cdot O\left(\text{SNR} \cdot N^{-\frac{1}{2}} d_h^{-\frac{1}{4}}\right) \cdot \log(d_h^{\frac{1}{2}})$$

$$= |\langle \boldsymbol{k}_+^{(T_1)}, \boldsymbol{k}_-^{(T_1)}\rangle| + O\left(\text{SNR} \cdot N^{\frac{1}{2}} d_h^{-\frac{1}{4}} \log(d_h^{\frac{1}{2}})\right)$$

$$= o(1),$$

where the first inequality is triangle inequality, the second inequality is by (214), the last equality is by $d_h = \widetilde{\Omega}\left(\max\{\text{SNR}^4, \text{SNR}^{-4}\} N^2 \epsilon^{-2}\right)$.

$$|\langle \boldsymbol{k}_+^{(t+1)}, \boldsymbol{k}_{n,i}^{(t+1)}\rangle| \le |\langle \boldsymbol{k}_+^{(T_1)}, \boldsymbol{k}_{n,i}^{(T_1)}\rangle| + \sum_{s=T_1}^{t} \left|\langle \boldsymbol{k}_+^{(s+1)}, \boldsymbol{k}_{n,i}^{(s+1)}\rangle - \langle \boldsymbol{k}_+^{(s)}, \boldsymbol{k}_{n,i}^{(s)}\rangle\right|$$

$$\le |\langle \boldsymbol{k}_+^{(T_1)}, \boldsymbol{k}_{n,i}^{(T_1)}\rangle|$$

$$+ \sum_{s=T_1}^{t} \left| \beta_{+,+}^{(s)} \langle \boldsymbol{q}_+^{(s)}, \boldsymbol{k}_{n,i}^{(s)}\rangle + \sum_{n'\in S_+} \sum_{l=2}^{M} \beta_{n',+,l}^{(s)} \langle \boldsymbol{q}_{n',l}^{(s)}, \boldsymbol{k}_{n,i}^{(s)}\rangle \right.$$

$$+ \beta_{n,i,+}^{(s)} \langle \boldsymbol{q}_+^{(s)}, \boldsymbol{k}_+^{(s)}\rangle + \beta_{n,i,-}^{(s)} \langle \boldsymbol{q}_-^{(s)}, \boldsymbol{k}_+^{(s)}\rangle + \sum_{n'=1}^{N} \sum_{l=2}^{M} \beta_{n,i,n',l}^{(s)} \langle \boldsymbol{q}_{n',l}^{(s)}, \boldsymbol{k}_+^{(s)}\rangle$$

$$+ \left(\beta_{+,+}^{(s)} \boldsymbol{q}_+^{(s)} + \sum_{n'\in S_+} \sum_{l=2}^{M} \beta_{n',+,l}^{(s)} \boldsymbol{q}_{n',l}^{(s)}\right)$$

$$\cdot \left(\beta_{n,i,+}^{(s)} \boldsymbol{q}_+^{(s)\top} + \beta_{n,i,-}^{(s)} \boldsymbol{q}_-^{(s)\top} + \sum_{n'=1}^{N} \sum_{l=2}^{M} \beta_{n,i,n',l}^{(s)} \boldsymbol{q}_{n',l}^{(s)\top}\right) \right|$$

$$\le |\langle \boldsymbol{k}_+^{(T_1)}, \boldsymbol{k}_{n,i}^{(T_1)}\rangle|$$

$$+ \sum_{s=T_1}^{t} |\beta_{+,+}^{(s)}||\langle \boldsymbol{q}_+^{(s)}, \boldsymbol{k}_{n,i}^{(s)}\rangle| + \sum_{l=2}^{M} \sum_{s=T_1}^{t} |\beta_{n,+,l}^{(s)}||\langle \boldsymbol{q}_{n,l}^{(s)}, \boldsymbol{k}_{n,i}^{(s)}\rangle|$$

$$+ \sum_{n'\in S_+ \wedge n'\ne n} \sum_{l=2}^{M} \sum_{s=T_1}^{t} |\beta_{n',+,l}^{(s)}||\langle \boldsymbol{q}_{n',l}^{(s)}, \boldsymbol{k}_{n,i}^{(s)}\rangle| + \sum_{s=T_1}^{t} |\beta_{n,i,+}^{(s)}||\langle \boldsymbol{q}_+^{(s)}, \boldsymbol{k}_+^{(s)}\rangle|$$

$$+ \sum_{s=T_1}^{t} |\beta_{n,i,-}^{(s)}||\langle \boldsymbol{q}_-^{(s)}, \boldsymbol{k}_+^{(s)}\rangle| + \sum_{l=2}^{M} \sum_{s=T_1}^{t} |\beta_{n,i,n,l}^{(s)}||\langle \boldsymbol{q}_{n,l}^{(s)}, \boldsymbol{k}_+^{(s)}\rangle|$$

$$+ \sum_{n'\ne n}^{N} \sum_{l=2}^{M} \sum_{s=T_1}^{t} |\beta_{n,i,n',l}^{(s)}||\langle \boldsymbol{q}_{n',l}^{(s)}, \boldsymbol{k}_+^{(s)}\rangle|$$

$$+ \{lower\ order\ term\}$$

$$= |\langle \boldsymbol{k}_+^{(T_1)}, \boldsymbol{k}_{n,i}^{(T_1)}\rangle|$$

$$+ O\left(N^{\frac{1}{2}} d_h^{-\frac{1}{4}}\right) \cdot \log(d_h^{\frac{1}{2}}) + M \cdot O\left(\text{SNR} \cdot N^{-\frac{1}{2}} d_h^{-\frac{1}{4}}\right) \cdot \log(d_h^{\frac{1}{2}})$$

$$+ N \cdot M \cdot O\left(\text{SNR} \cdot N^{-\frac{1}{2}} d_h^{-\frac{1}{4}}\right) \cdot o(1) + O\left(N^{\frac{1}{2}} d_h^{-\frac{1}{4}}\right) \cdot \log(d_h^{\frac{1}{2}})$$

$$+ M \cdot O\left(d_h^{-\frac{1}{4}}\right) \cdot \log(d_h^{\frac{1}{2}}) + N \cdot M \cdot O\left(d^{-\frac{1}{2}} d_h^{-\frac{1}{4}} \log(6N^2 M^2/\delta)\right) \cdot \log(d_h^{\frac{1}{2}})$$

$$= |\langle \boldsymbol{k}_+^{(T_1)}, \boldsymbol{k}_{n,i}^{(T_1)} \rangle| + o\left(\text{SNR} \cdot N^{\frac{1}{2}} d_h^{-\frac{1}{4}}\right) + O\left(N^{\frac{1}{2}} d_h^{-\frac{1}{4}} \log(d_h^{\frac{1}{2}})\right)$$

$$+ O\left(N d^{-\frac{1}{2}} d_h^{-\frac{1}{4}} \log(6N^2 M^2/\delta) \cdot \log(d_h^{\frac{1}{2}})\right)$$

$$= o(1)$$

where the first inequality is triangle inequality, the second inequality is by (209), the last equality is by $d_h = \widetilde{\Omega}\left(\max\{\text{SNR}^4, \text{SNR}^{-4}\} N^2 \epsilon^{-2}\right)$ and $d = \widetilde{\Omega}\left(\epsilon^{-2} N^2 d_h\right)$. Similarly, we have $|\langle \boldsymbol{k}_-^{(t+1)}, \boldsymbol{k}_{n,i}^{(t+1)} \rangle| = o(1)$.

$$|\langle \boldsymbol{k}_{n,i}^{(t+1)}, \boldsymbol{k}_{n,j}^{(t+1)} \rangle| \le |\langle \boldsymbol{k}_{n,i}^{(T_1)}, \boldsymbol{k}_{n,j}^{(T_1)} \rangle| + \sum_{s=T_1}^{t} \left| \langle \boldsymbol{k}_{n,i}^{(s+1)}, \boldsymbol{k}_{n,j}^{(s+1)} \rangle - \langle \boldsymbol{k}_{n,i}^{(s)}, \boldsymbol{k}_{n,j}^{(s)} \rangle \right|$$

$$\le |\langle \boldsymbol{k}_{n,i}^{(T_1)}, \boldsymbol{k}_{n,j}^{(T_1)} \rangle|$$

$$+ \sum_{s=T_1}^{t} \left| \beta_{n,i,+}^{(s)} \langle \boldsymbol{q}_+^{(s)}, \boldsymbol{k}_{n,j}^{(s)} \rangle + \beta_{n,i,-}^{(s)} \langle \boldsymbol{q}_-^{(s)}, \boldsymbol{k}_{n,j}^{(s)} \rangle + \sum_{n'=1}^{N} \sum_{l=2}^{M} \beta_{n,i,n',l}^{(s)} \langle \boldsymbol{q}_{n',l}^{(s)}, \boldsymbol{k}_{n,j}^{(s)} \rangle \right.$$

$$+ \beta_{n,j,+}^{(s)} \langle \boldsymbol{q}_+^{(s)}, \boldsymbol{k}_{n,i}^{(s)} \rangle + \beta_{n,j,-}^{(s)} \langle \boldsymbol{q}_-^{(s)}, \boldsymbol{k}_{n,i}^{(s)} \rangle + \sum_{n'=1}^{N} \sum_{l=2}^{M} \beta_{n,j,n',l}^{(s)} \langle \boldsymbol{q}_{n',l}^{(s)}, \boldsymbol{k}_{n,i}^{(s)} \rangle$$

$$+ \left( \beta_{n,i,+}^{(s)} \boldsymbol{q}_+^{(s)} + \beta_{n,i,-}^{(s)} \boldsymbol{q}_-^{(s)} + \sum_{n'=1}^{N} \sum_{l=2}^{M} \beta_{n,i,n',l}^{(s)} \boldsymbol{q}_{n',l}^{(s)} \right)$$

$$\left. \cdot \left( \beta_{n,j,+}^{(s)} \boldsymbol{q}_+^{(s)\top} + \beta_{n,j,-}^{(s)} \boldsymbol{q}_-^{(s)\top} + \sum_{n'=1}^{N} \sum_{l=2}^{M} \beta_{n,j,n',l}^{(s)} \boldsymbol{q}_{n',l}^{(s)\top} \right) \right|$$

$$\le |\langle \boldsymbol{k}_{n,i}^{(T_1)}, \boldsymbol{k}_{n,j}^{(T_1)} \rangle|$$

$$+ \sum_{s=T_1}^{t} |\beta_{n,i,+}^{(s)}| |\langle \boldsymbol{q}_+^{(s)}, \boldsymbol{k}_{n,j}^{(s)} \rangle| + \sum_{s=T_1}^{t} |\beta_{n,i,-}^{(s)}| |\langle \boldsymbol{q}_-^{(s)}, \boldsymbol{k}_{n,j}^{(s)} \rangle| + \sum_{l=2}^{M} \sum_{s=T_1}^{t} |\beta_{n,i,n,l}^{(s)}| |\langle \boldsymbol{q}_{n,l}^{(s)}, \boldsymbol{k}_{n,j}^{(s)} \rangle|$$

$$+ \sum_{n' \ne n}^{N} \sum_{l=2}^{M} \sum_{s=T_1}^{t} |\beta_{n,i,n',l}^{(s)}| |\langle \boldsymbol{q}_{n',l}^{(s)}, \boldsymbol{k}_{n,j}^{(s)} \rangle| + \sum_{s=T_1}^{t} |\beta_{n,j,+}^{(s)}| |\langle \boldsymbol{q}_+^{(s)}, \boldsymbol{k}_{n,i}^{(s)} \rangle| + \sum_{s=T_1}^{t} |\beta_{n,j,-}^{(s)}| |\langle \boldsymbol{q}_-^{(s)}, \boldsymbol{k}_{n,i}^{(s)} \rangle|$$

$$+ \sum_{l=2}^{M} \sum_{s=T_1}^{t} |\beta_{n,j,n,l}^{(s)}| |\langle \boldsymbol{q}_{n,l}^{(s)}, \boldsymbol{k}_{n,i}^{(s)} \rangle| + \sum_{n' \ne n}^{N} \sum_{l=2}^{M} \sum_{s=T_1}^{t} |\beta_{n,j,n',l}^{(s)}| |\langle \boldsymbol{q}_{n',l}^{(s)}, \boldsymbol{k}_{n,i}^{(s)} \rangle|$$

$$+ \{lower\ order\ term\}$$

$$= |\langle \boldsymbol{k}_{n,i}^{(T_1)}, \boldsymbol{k}_{n,j}^{(T_1)} \rangle|$$

$$+ O\left(N^{\frac{1}{2}} d_h^{-\frac{1}{4}}\right) \cdot \log(d_h^{\frac{1}{2}}) + M \cdot O\left(d_h^{-\frac{1}{4}}\right) \cdot \log(d_h^{\frac{1}{2}})$$

$$+ N \cdot M \cdot O\left(d^{-\frac{1}{2}} d_h^{-\frac{1}{4}} \log(6N^2 M^2/\delta)\right) \cdot o(1)$$

$$= |\langle \boldsymbol{k}_{n,i}^{(T_1)}, \boldsymbol{k}_{n,j}^{(T_1)} \rangle| + O\left(N^{\frac{1}{2}} d_h^{-\frac{1}{4}} \log(d_h^{\frac{1}{2}})\right) + o\left(N d^{-\frac{1}{2}} d_h^{-\frac{1}{4}} \log(6N^2 M^2/\delta)\right)$$

$$= o(1)$$

for $i, j \in [M]\backslash\{1\}, i \ne j, n \in [N]$. The first inequality is triangle inequality, the second inequality is by (211), the last equality is by $d_h = \widetilde{\Omega}\left(\max\{\text{SNR}^4, \text{SNR}^{-4}\} N^2 \epsilon^{-2}\right)$ and $d = \widetilde{\Omega}\left(\epsilon^{-2} N^2 d_h\right)$.

$$|\langle \boldsymbol{k}_{n,i}^{(t+1)}, \boldsymbol{k}_{\overline{n},j}^{(t+1)} \rangle| \le |\langle \boldsymbol{k}_{n,i}^{(T_1)}, \boldsymbol{k}_{\overline{n},j}^{(T_1)} \rangle| + \sum_{s=T_1}^{t} \left| \langle \boldsymbol{k}_{n,i}^{(s+1)}, \boldsymbol{k}_{\overline{n},j}^{(s+1)} \rangle - \langle \boldsymbol{k}_{n,i}^{(s)}, \boldsymbol{k}_{\overline{n},j}^{(s)} \rangle \right|$$

$$\le |\langle \boldsymbol{k}_{n,i}^{(T_1)}, \boldsymbol{k}_{\overline{n},j}^{(T_1)} \rangle|$$

$$+ \sum_{s=T_1}^{t} \left| \beta_{n,i,+}^{(s)} \langle \boldsymbol{q}_+^{(s)}, \boldsymbol{k}_{\overline{n},j}^{(s)} \rangle + \beta_{n,i,-}^{(s)} \langle \boldsymbol{q}_-^{(s)}, \boldsymbol{k}_{\overline{n},j}^{(s)} \rangle + \sum_{n'=1}^{N} \sum_{l=2}^{M} \beta_{n,i,n',l}^{(s)} \langle \boldsymbol{q}_{n',l}^{(s)}, \boldsymbol{k}_{\overline{n},j}^{(s)} \rangle \right.$$

$$+ \beta_{\overline{n},j,+}^{(s)} \langle \boldsymbol{q}_+^{(s)}, \boldsymbol{k}_{n,i}^{(s)} \rangle + \beta_{\overline{n},j,-}^{(s)} \langle \boldsymbol{q}_-^{(s)}, \boldsymbol{k}_{n,i}^{(s)} \rangle + \sum_{n'=1}^{N} \sum_{l=2}^{M} \beta_{\overline{n},j,n',l}^{(s)} \langle \boldsymbol{q}_{n',l}^{(s)}, \boldsymbol{k}_{n,i}^{(s)} \rangle$$

$$+ \left( \beta_{n,i,+}^{(s)} \boldsymbol{q}_+^{(s)} + \beta_{n,i,-}^{(s)} \boldsymbol{q}_-^{(s)} + \sum_{n'=1}^{N} \sum_{l=2}^{M} \beta_{n,i,n',l}^{(s)} \boldsymbol{q}_{n',l}^{(s)} \right)$$

$$\left. \cdot \left( \beta_{\overline{n},j,+}^{(s)} \boldsymbol{q}_+^{(s)\top} + \beta_{\overline{n},j,-}^{(s)} \boldsymbol{q}_-^{(s)\top} + \sum_{n'=1}^{N} \sum_{l=2}^{M} \beta_{\overline{n},j,n',l}^{(s)} \boldsymbol{q}_{n',l}^{(s)\top} \right) \right|$$

$$\leq |\langle \boldsymbol{k}_{n,i}^{(T_1)}, \boldsymbol{k}_{\overline{n},j}^{(T_1)} \rangle|$$

$$+ \sum_{s=T_1}^{t} |\beta_{n,i,+}^{(s)}| |\langle \boldsymbol{q}_+^{(s)}, \boldsymbol{k}_{\overline{n},j}^{(s)} \rangle| + \sum_{s=T_1}^{t} |\beta_{n,i,-}^{(s)}| |\langle \boldsymbol{q}_-^{(s)}, \boldsymbol{k}_{\overline{n},j}^{(s)} \rangle| + \sum_{l=2}^{M} \sum_{s=T_1}^{t} |\beta_{n,i,\overline{n},l}^{(s)}| |\langle \boldsymbol{q}_{\overline{n},l}^{(s)}, \boldsymbol{k}_{\overline{n},j}^{(s)} \rangle|$$

$$+ \sum_{l=2}^{M} \sum_{s=T_1}^{t} |\beta_{n,i,n,l}^{(s)}| |\langle \boldsymbol{q}_{n,l}^{(s)}, \boldsymbol{k}_{\overline{n},j}^{(s)} \rangle| + \sum_{n' \neq n \wedge n' \overline{n}} \sum_{l=2}^{M} \sum_{s=T_1}^{t} |\beta_{n,i,n',l}^{(s)}| |\langle \boldsymbol{q}_{n',l}^{(s)}, \boldsymbol{k}_{\overline{n},j}^{(s)} \rangle|$$

$$+ \sum_{s=T_1}^{t} |\beta_{\overline{n},j,+}^{(s)}| |\langle \boldsymbol{q}_+^{(s)}, \boldsymbol{k}_{n,i}^{(s)} \rangle| + \sum_{s=T_1}^{t} |\beta_{\overline{n},j,-}^{(s)}| |\langle \boldsymbol{q}_-^{(s)}, \boldsymbol{k}_{n,i}^{(s)} \rangle| + \sum_{l=2}^{M} \sum_{s=T_1}^{t} |\beta_{\overline{n},j,n,l}^{(s)}| |\langle \boldsymbol{q}_{n,l}^{(s)}, \boldsymbol{k}_{n,i}^{(s)} \rangle|$$

$$+ \sum_{l=2}^{M} \sum_{s=T_1}^{t} |\beta_{\overline{n},j,\overline{n},l}^{(s)}| |\langle \boldsymbol{q}_{\overline{n},l}^{(s)}, \boldsymbol{k}_{n,i}^{(s)} \rangle| + \sum_{n' \neq n \wedge n' \overline{n}} \sum_{l=2}^{M} \sum_{s=T_1}^{t} |\beta_{\overline{n},j,n',l}^{(s)}| |\langle \boldsymbol{q}_{n',l}^{(s)}, \boldsymbol{k}_{n,i}^{(s)} \rangle|$$

$$+ \{lower\ order\ term\}$$

$$= |\langle \boldsymbol{k}_{n,i}^{(T_1)}, \boldsymbol{k}_{\overline{n},j}^{(T_1)} \rangle|$$

$$+ O\left( N^{\frac{1}{2}} d_h^{-\frac{1}{4}} \right) \cdot \log(d_h^{\frac{1}{2}}) + M \cdot O\left( d^{-\frac{1}{2}} d_h^{-\frac{1}{4}} \log(6N^2 M^2/\delta) \right) \cdot \log(d_h^{\frac{1}{2}})$$

$$+ M \cdot O\left( d_h^{-\frac{1}{4}} \right) \cdot o(1) + N \cdot M \cdot O\left( d^{-\frac{1}{2}} d_h^{-\frac{1}{4}} \log(6N^2 M^2/\delta) \right) \cdot o(1)$$

$$= |\langle \boldsymbol{k}_{n,i}^{(T_1)}, \boldsymbol{k}_{\overline{n},j}^{(T_1)} \rangle|$$

$$+ O\left( N^{\frac{1}{2}} d_h^{-\frac{1}{4}} \log(d_h^{\frac{1}{2}}) \right) + o\left( N d^{-\frac{1}{2}} d_h^{-\frac{1}{4}} \log(6N^2 M^2/\delta) \right)$$

$$= o(1)$$

for $i, j \in [M] \backslash \{1\}, n, \overline{n} \in [N], n \neq \overline{n}$. The first inequality is triangle inequality, the second inequality is by (212), the last equality is by $d_h = \widetilde{\Omega}\left( \max\{\text{SNR}^4, \text{SNR}^{-4}\} N^2 \epsilon^{-2} \right)$ and $d = \widetilde{\Omega}\left( \epsilon^{-2} N^2 d_h \right)$.

### F.9   Upper Bounds of $\langle$ q, k $\rangle$

In order to give the upper bounds for $\langle \boldsymbol{q}, \boldsymbol{k} \rangle$ in stage III, we need to give the upper bounds of $\alpha$ and $\beta$ based on the equations in F.1. The main difference between this subsection and F.6 is that the bounds of $|V_\pm|, |V_{n,i}|$ is $\log\left( O(\frac{1}{\epsilon}) \right)$ in this subsection, while the bounds of $|V_\pm|, |V_{n,i}|$ is $\log\left( O(\frac{1}{\epsilon}) \right)$ in F.6, resulting in different bounds for $\alpha$ and $\beta$. Now we take $\alpha_{+,+}^{(s)}$ as an example

$$\alpha_{+,+}^{(s)} = \frac{\eta}{NM} \sum_{n \in S_+} -\ell_n'^{(s)} \|\boldsymbol{\mu}\|_2^2$$

$$\cdot \left( V_+^{(s)} \left( \frac{\exp(\langle \boldsymbol{q}_+^{(s)}, \boldsymbol{k}_+^{(s)} \rangle)}{\exp(\langle \boldsymbol{q}_+^{(s)}, \boldsymbol{k}_+^{(s)} \rangle) + \sum_{j=2}^{M} \exp(\langle \boldsymbol{q}_+^{(s)}, \boldsymbol{k}_{n,j}^{(s)} \rangle)} \right. \right.$$

$$
- (\frac{\exp(\langle \boldsymbol{q}_+^{(s)}, \boldsymbol{k}_+^{(s)} \rangle)}{\exp(\langle \boldsymbol{q}_+^{(s)}, \boldsymbol{k}_+^{(s)} \rangle) + \sum\limits_{j=2}^{M} \exp(\langle \boldsymbol{q}_+^{(s)}, \boldsymbol{k}_{n,j}^{(s)} \rangle)})^2)
$$

$$
- \sum_{i=2}^{M} (V_{n,i}^{(s)} \cdot \frac{\exp(\langle \boldsymbol{q}_+^{(s)}, \boldsymbol{k}_+^{(s)} \rangle)}{\exp(\langle \boldsymbol{q}_+^{(s)}, \boldsymbol{k}_+^{(s)} \rangle) + \sum\limits_{j=2}^{M} \exp(\langle \boldsymbol{q}_+^{(s)}, \boldsymbol{k}_{n,j}^{(s)} \rangle)}
$$

$$
\cdot \frac{\exp(\langle \boldsymbol{q}_+^{(s)}, \boldsymbol{k}_{n,i}^{(s)} \rangle)}{\exp(\langle \boldsymbol{q}_+^{(s)}, \boldsymbol{k}_+^{(s)} \rangle) + \sum\limits_{j=2}^{M} \exp(\langle \boldsymbol{q}_+^{(s)}, \boldsymbol{k}_{n,j}^{(s)} \rangle)}))
$$

$$
= \frac{\eta}{NM} \sum_{n \in S_+} -\ell_n'^{(s)} \|\boldsymbol{\mu}\|_2^2 \frac{\exp(\langle \boldsymbol{q}_+^{(s)}, \boldsymbol{k}_+^{(s)} \rangle)}{\exp(\langle \boldsymbol{q}_+^{(s)}, \boldsymbol{k}_+^{(s)} \rangle) + \sum\limits_{j=2}^{M} \exp(\langle \boldsymbol{q}_+^{(s)}, \boldsymbol{k}_{n,j}^{(s)} \rangle)}
$$

$$
\cdot \left( V_+^{(s)} \cdot \frac{\sum\limits_{j=2}^{M} \exp(\langle \boldsymbol{q}_+^{(s)}, \boldsymbol{k}_{n,j}^{(s)} \rangle)}{\exp(\langle \boldsymbol{q}_+^{(s)}, \boldsymbol{k}_+^{(s)} \rangle) + \sum\limits_{j=2}^{M} \exp(\langle \boldsymbol{q}_+^{(s)}, \boldsymbol{k}_{n,j}^{(s)} \rangle)} \right.
$$

$$
\left. - \sum_{i=2}^{M} V_{n,i}^{(s)} \cdot \frac{\exp(\langle \boldsymbol{q}_+^{(s)}, \boldsymbol{k}_{n,i}^{(s)} \rangle)}{\exp(\langle \boldsymbol{q}_+^{(s)}, \boldsymbol{k}_+^{(s)} \rangle) + \sum\limits_{j=2}^{M} \exp(\langle \boldsymbol{q}_+^{(s)}, \boldsymbol{k}_{n,j}^{(s)} \rangle)} \right)
$$

$$
\leq \frac{\eta}{NM} \sum_{n \in S_+} \|\boldsymbol{\mu}\|_2^2 \cdot \left( V_+^{(s)} \cdot \frac{\sum\limits_{j=2}^{M} \exp(\langle \boldsymbol{q}_+^{(s)}, \boldsymbol{k}_{n,j}^{(s)} \rangle)}{\exp(\langle \boldsymbol{q}_+^{(s)}, \boldsymbol{k}_+^{(s)} \rangle) + \sum\limits_{j=2}^{M} \exp(\langle \boldsymbol{q}_+^{(s)}, \boldsymbol{k}_{n,j}^{(s)} \rangle)} \right.
$$

$$
\left. + \max_i |V_{n,i}^{(s)}| \cdot \frac{\sum\limits_{j=2}^{M} \exp(\langle \boldsymbol{q}_+^{(s)}, \boldsymbol{k}_{n,j}^{(s)} \rangle)}{\exp(\langle \boldsymbol{q}_+^{(s)}, \boldsymbol{k}_+^{(s)} \rangle) + \sum\limits_{j=2}^{M} \exp(\langle \boldsymbol{q}_+^{(s)}, \boldsymbol{k}_{n,j}^{(s)} \rangle)} \right)
$$

$$
\leq \frac{\eta}{NM} \cdot \frac{3N}{4} \cdot \|\boldsymbol{\mu}\|_2^2 \cdot \left( V_+^{(s)} \cdot \frac{C}{\exp(\langle \boldsymbol{q}_+^{(s)}, \boldsymbol{k}_+^{(s)} \rangle)} + \max_i |V_{n,i}^{(s)}| \cdot \frac{C}{\exp(\langle \boldsymbol{q}_+^{(s)}, \boldsymbol{k}_+^{(s)} \rangle)} \right)
$$

$$
\leq \frac{\eta C_9 \|\boldsymbol{\mu}\|_2^2 \log \left( O(\frac{1}{\epsilon}) \right)}{\exp(\langle \boldsymbol{q}_+^{(s)}, \boldsymbol{k}_+^{(s)} \rangle)},
$$

where the first inequality is by $-\ell_n'^{(s)} \leq 1$ and $softmax(\langle \boldsymbol{q}_+^{(s)}, \boldsymbol{k}_+^{(s)} \rangle) \leq 1$. For the second inequality, we first consider $\frac{\sum\limits_{j=2}^{M} \exp(\langle \boldsymbol{q}_+^{(s)}, \boldsymbol{k}_{n,j}^{(s)} \rangle)}{\exp(\langle \boldsymbol{q}_+^{(s)}, \boldsymbol{k}_+^{(s)} \rangle) + \sum\limits_{j=2}^{M} \exp(\langle \boldsymbol{q}_+^{(s)}, \boldsymbol{k}_{n,j}^{(s)} \rangle)} \leq \frac{\sum\limits_{j=2}^{M} \exp(\langle \boldsymbol{q}_+^{(s)}, \boldsymbol{k}_{n,j}^{(s)} \rangle)}{\exp(\langle \boldsymbol{q}_+^{(s)}, \boldsymbol{k}_+^{(s)} \rangle)}$, then by the monotonicity of $\langle \boldsymbol{q}_+^{(s)}, \boldsymbol{k}_{n,j}^{(s)} \rangle$ and $\langle \boldsymbol{q}_+^{(T_1)}, \boldsymbol{k}_{n,j}^{(T_1)} \rangle = o(1)$ we have $\sum\limits_{j=2}^{M} \exp(\langle \boldsymbol{q}_+^{(s)}, \boldsymbol{k}_{n,j}^{(s)} \rangle) \leq C$ for $t \in [T_1, T_3]$. The last inequality is by $V_+^{(s)}, |V_{n,i}^{(s)}| \leq 2 \log \left( O(\frac{1}{\epsilon}) \right)$ for $t \in [T_2, T_3]$ and absorbing the constant factors. Similar to F.6, we can give the bounds for the other $\alpha$ and $\beta$ as follows:

$$
\alpha_{-,-}^{(s)} \leq \frac{\eta C_9 \|\boldsymbol{\mu}\|_2^2 \log \left( O(\frac{1}{\epsilon}) \right)}{\exp(\langle \boldsymbol{q}_-^{(s)}, \boldsymbol{k}_-^{(s)} \rangle)},
$$

$$
\beta_{+,+}^{(s)} \leq \frac{\eta C_9 \|\boldsymbol{\mu}\|_2^2 \log \left( O(\frac{1}{\epsilon}) \right)}{\exp(\langle \boldsymbol{q}_+^{(s)}, \boldsymbol{k}_+^{(s)} \rangle)},
$$

$$\beta_{-,-}^{(s)} \le \frac{\eta C_9 \|\boldsymbol{\mu}\|_2^2 \log\left(O(\frac{1}{\epsilon})\right)}{\exp(\langle \boldsymbol{q}_-^{(s)}, \boldsymbol{k}_-^{(s)} \rangle)}.$$

$$\alpha_{n,+,j}^{(s)} \ge -\frac{\eta C_9 \|\boldsymbol{\mu}\|_2^2 \log\left(O(\frac{1}{\epsilon})\right)}{N} \cdot \exp(\langle \boldsymbol{q}_+^{(s)}, \boldsymbol{k}_{n,j}^{(s)} \rangle),$$

$$\alpha_{n,-,j}^{(s)} \ge -\frac{\eta C_9 \|\boldsymbol{\mu}\|_2^2 \log\left(O(\frac{1}{\epsilon})\right)}{N} \cdot \exp(\langle \boldsymbol{q}_-^{(s)}, \boldsymbol{k}_{n,j}^{(s)} \rangle),$$

$$\beta_{n,+,i}^{(s)} \le \frac{\eta C_9 \|\boldsymbol{\mu}\|_2^2 \log\left(O(\frac{1}{\epsilon})\right)}{N \exp(\langle \boldsymbol{q}_{n,i}^{(s)}, \boldsymbol{k}_+^{(s)} \rangle)},$$

$$\beta_{n,-,i}^{(s)} \le \frac{\eta C_9 \|\boldsymbol{\mu}\|_2^2 \log\left(O(\frac{1}{\epsilon})\right)}{N \exp(\langle \boldsymbol{q}_{n,i}^{(s)}, \boldsymbol{k}_-^{(s)} \rangle)},$$

$$\alpha_{n,i,+}^{(s)} \le \frac{\eta C_9 \sigma_p^2 d \log\left(O(\frac{1}{\epsilon})\right)}{N \exp(\langle \boldsymbol{q}_{n,i}^{(s)}, \boldsymbol{k}_+^{(s)} \rangle)},$$

$$\alpha_{n,i,-}^{(s)} \le \frac{\eta C_9 \sigma_p^2 d \log\left(O(\frac{1}{\epsilon})\right)}{N \exp(\langle \boldsymbol{q}_{n,i}^{(s)}, \boldsymbol{k}_-^{(s)} \rangle)},$$

$$\beta_{n,i,+}^{(s)} \ge -\frac{\eta C_9 \sigma_p^2 d \log\left(O(\frac{1}{\epsilon})\right)}{N} \exp(\langle \boldsymbol{q}_+^{(s)}, \boldsymbol{k}_{n,i}^{(s)} \rangle),$$

$$\beta_{n,i,-}^{(s)} \ge -\frac{\eta C_9 \sigma_p^2 d \log\left(O(\frac{1}{\epsilon})\right)}{N} \exp(\langle \boldsymbol{q}_-^{(s)}, \boldsymbol{k}_{n,i}^{(s)} \rangle),$$

$$\alpha_{n,i,n,j}^{(s)} \ge -\frac{\eta C_9 \sigma_p^2 d \log\left(O(\frac{1}{\epsilon})\right)}{N} \exp(\langle \boldsymbol{q}_{n,i}^{(s)}, \boldsymbol{k}_{n,j}^{(s)} \rangle),$$

$$\beta_{n,j,n,i}^{(s)} \ge -\frac{\eta C_9 \sigma_p^2 d \log\left(O(\frac{1}{\epsilon})\right)}{N} \exp(\langle \boldsymbol{q}_{n,i}^{(s)}, \boldsymbol{k}_{n,j}^{(s)} \rangle).$$

Similar to F.5, we apply the bounds of $\alpha$ and $\beta$ above to give the upper bounds for the dynamics $\langle \boldsymbol{q}, \boldsymbol{k} \rangle$.

$$\langle \boldsymbol{q}_+^{(s+1)}, \boldsymbol{k}_+^{(s+1)} \rangle - \langle \boldsymbol{q}_+^{(s)}, \boldsymbol{k}_+^{(s)} \rangle$$

$$= \alpha_{+,+}^{(s)} \|\boldsymbol{k}_+^{(s)}\|_2^2 + \sum_{n \in S_+} \sum_{i=2}^{M} \alpha_{n,+,i}^{(s)} \langle \boldsymbol{k}_+^{(s)}, \boldsymbol{k}_{n,i}^{(s)} \rangle$$

$$+ \beta_{+,+}^{(s)} \|\boldsymbol{q}_+^{(s)}\|_2^2 + \sum_{n \in S_+} \sum_{i=2}^{M} \beta_{n,+,i}^{(s)} \langle \boldsymbol{q}_+^{(s)}, \boldsymbol{q}_{n,i}^{(s)} \rangle$$

$$+ \left( \alpha_{+,+}^{(s)} \boldsymbol{k}_+^{(s)} + \sum_{n \in S_+} \sum_{i=2}^{M} \alpha_{n,+,i}^{(s)} \boldsymbol{k}_{n,i}^{(s)} \right)$$

$$\cdot \left( \beta_{+,+}^{(s)} \boldsymbol{q}_+^{(s)\top} + \sum_{n \in S_+} \sum_{i=2}^{M} \beta_{n,+,i}^{(s)} \boldsymbol{q}_{n,i}^{(s)\top} \right) \tag{271}$$

$$= \alpha_{+,+}^{(s)} \|\boldsymbol{k}_+^{(s)}\|_2^2 + \beta_{+,+}^{(s)} \|\boldsymbol{q}_+^{(s)}\|_2^2 + \{lower\ order\ term\}$$

$$\leq \frac{2\eta C_9 \|\boldsymbol{\mu}\|_2^2 \log \left( O(\frac{1}{\epsilon}) \right)}{\exp(\langle \boldsymbol{q}_+^{(s)}, \boldsymbol{k}_+^{(s)} \rangle)} \cdot \Theta(\|\boldsymbol{\mu}\|_2^2 \sigma_h^2 d_h) + \{lower\ order\ term\}$$

$$\leq \frac{\eta C_{10} \|\boldsymbol{\mu}\|_2^4 \sigma_h^2 d_h \log \left( O(\frac{1}{\epsilon}) \right)}{\exp(\langle \boldsymbol{q}_+^{(s)}, \boldsymbol{k}_+^{(s)} \rangle)},$$

similarly, we have

$$\langle \boldsymbol{q}_-^{(s+1)}, \boldsymbol{k}_-^{(s+1)} \rangle - \langle \boldsymbol{q}_-^{(s)}, \boldsymbol{k}_-^{(s)} \rangle \leq \frac{\eta C_{10} \|\boldsymbol{\mu}\|_2^4 \sigma_h^2 d_h \log \left( O(\frac{1}{\epsilon}) \right)}{\exp(\langle \boldsymbol{q}_-^{(s)}, \boldsymbol{k}_-^{(s)} \rangle)}. \tag{272}$$

$$\langle \boldsymbol{q}_+^{(s+1)}, \boldsymbol{k}_{n,j}^{(s+1)} \rangle - \langle \boldsymbol{q}_+^{(s)}, \boldsymbol{k}_{n,j}^{(s)} \rangle$$

$$= \alpha_{+,+}^{(s)} \langle \boldsymbol{k}_+^{(s)}, \boldsymbol{k}_{n,j}^{(s)} \rangle + \sum_{n' \in S_+} \sum_{l=2}^{M} \alpha_{n',+,l}^{(s)} \langle \boldsymbol{k}_{n,j}^{(s)}, \boldsymbol{k}_{n',l}^{(s)} \rangle$$

$$+ \beta_{n,j,+}^{(s)} \|\boldsymbol{q}_+^{(s)}\|_2^2 + \beta_{n,j,-}^{(s)} \langle \boldsymbol{q}_+^{(s)}, \boldsymbol{q}_-^{(s)} \rangle + \sum_{n'=1}^{N} \sum_{l=2}^{M} \beta_{n,j,n',l}^{(s)} \langle \boldsymbol{q}_+^{(s)}, \boldsymbol{q}_{n',l}^{(s)} \rangle$$

$$+ \left( \alpha_{+,+}^{(s)} \boldsymbol{k}_+^{(s)} + \sum_{n' \in S_+} \sum_{l=2}^{M} \alpha_{n',+,l}^{(s)} \boldsymbol{k}_{n',l}^{(s)} \right)$$

$$\cdot \left( \beta_{n,j,+}^{(s)} \boldsymbol{q}_+^{(s)\top} + \beta_{n,j,-}^{(s)} \boldsymbol{q}_-^{(s)\top} + \sum_{n'=1}^{N} \sum_{l=2}^{M} \beta_{n,j,n',l}^{(s)} \boldsymbol{q}_{n',l}^{(s)\top} \right) \tag{273}$$

$$= \alpha_{n,+,j}^{(s)} \|\boldsymbol{k}_{n,j}^{(s)}\|_2^2 + \beta_{n,j,+}^{(s)} \|\boldsymbol{q}_+^{(s)}\|_2^2 + \{lower\ order\ term\}$$

$$\geq -\frac{\eta C_9 \|\boldsymbol{\mu}\|_2^2 \log \left( O(\frac{1}{\epsilon}) \right)}{N} \cdot \exp(\langle \boldsymbol{q}_+^{(s)}, \boldsymbol{k}_{n,j}^{(s)} \rangle) \cdot \Theta \left( \sigma_p^2 \sigma_h^2 d d_h \right)$$

$$- \frac{\eta C_9 \sigma_p^2 d \log \left( O(\frac{1}{\epsilon}) \right)}{N} \exp(\langle \boldsymbol{q}_+^{(s)}, \boldsymbol{k}_{n,j}^{(s)} \rangle) \cdot \Theta \left( \|\boldsymbol{\mu}\|_2^2 \sigma_h^2 d_h \right)$$

$$+ \{lower\ order\ term\}$$

$$\geq -\frac{\eta C_{10} \sigma_p^2 d \|\boldsymbol{\mu}\|_2^2 \sigma_h^2 d_h \log \left( O(\frac{1}{\epsilon}) \right)}{N} \cdot \exp(\langle \boldsymbol{q}_+^{(s)}, \boldsymbol{k}_{n,j}^{(s)} \rangle),$$

similarly, we have

$$\langle \boldsymbol{q}_-^{(s+1)}, \boldsymbol{k}_{n,j}^{(s+1)} \rangle - \langle \boldsymbol{q}_-^{(s)}, \boldsymbol{k}_{n,j}^{(s)} \rangle \geq -\frac{\eta C_{10} \sigma_p^2 d \|\boldsymbol{\mu}\|_2^2 \sigma_h^2 d_h \log \left( O(\frac{1}{\epsilon}) \right)}{N} \cdot \exp(\langle \boldsymbol{q}_-^{(s)}, \boldsymbol{k}_{n,j}^{(s)} \rangle). \tag{274}$$

$$\langle \boldsymbol{q}_{n,i}^{(s+1)}, \boldsymbol{k}_+^{(s+1)} \rangle - \langle \boldsymbol{q}_{n,i}^{(s)}, \boldsymbol{k}_+^{(s)} \rangle$$

$$= \alpha_{n,i,+}^{(s)} \|\boldsymbol{k}_+^{(s)}\|_2^2 + \alpha_{n,i,-}^{(s)} \langle \boldsymbol{k}_+^{(s)}, \boldsymbol{k}_-^{(s)} \rangle + \sum_{n'=1}^{N} \sum_{l=2}^{M} \alpha_{n,i,n',l}^{(s)} \langle \boldsymbol{k}_+^{(s)}, \boldsymbol{k}_{n',l}^{(s)} \rangle$$

$$+ \beta_{+,+}^{(s)} \langle \boldsymbol{q}_+^{(s)}, \boldsymbol{q}_{n,i}^{(s)} \rangle + \sum_{n' \in S_+} \sum_{l=2}^{M} \beta_{n',+,l}^{(s)} \langle \boldsymbol{q}_{n,i}^{(s)}, \boldsymbol{q}_{n',l}^{(s)} \rangle$$

$$+ \left( \alpha_{n,i,+}^{(s)} \boldsymbol{k}_+^{(s)} + \alpha_{n,i,-}^{(s)} \boldsymbol{k}_-^{(s)} + \sum_{n'=1}^{N} \sum_{l=2}^{M} \alpha_{n,i,n',l}^{(s)} \boldsymbol{k}_{n',l}^{(s)} \right)$$

$$\cdot \left( \beta_{+,+}^{(s)} \boldsymbol{q}_+^{(s)\top} + \sum_{n' \in S_+} \sum_{l=2}^{M} \beta_{n',+,l}^{(s)} \boldsymbol{q}_{n',l}^{(s)\top} \right) \tag{275}$$

$$= \alpha_{n,i,+}^{(s)} \|\boldsymbol{k}_+^{(s)}\|_2^2 + \beta_{n,+,i}^{(s)} \|\boldsymbol{q}_{n,i}^{(s)}\|_2^2 + \{lower\ order\ term\}$$

$$\leq \frac{\eta C_9 \sigma_p^2 d \log \left( O(\frac{1}{\epsilon}) \right)}{N \exp(\langle \boldsymbol{q}_{n,i}^{(s)}, \boldsymbol{k}_+^{(s)} \rangle)} \cdot \Theta \left( \|\boldsymbol{\mu}\|_2^2 \sigma_h^2 d_h \right)$$

$$+ \frac{\eta C_9 \|\boldsymbol{\mu}\|_2^2 \log \left( O(\frac{1}{\epsilon}) \right)}{N \exp(\langle \boldsymbol{q}_{n,i}^{(s)}, \boldsymbol{k}_+^{(s)} \rangle)} \cdot \Theta \left( \sigma_p^2 \sigma_h^2 d d_h \right)$$

$$+ \{lower\ order\ term\}$$

$$\leq \frac{\eta C_{10} \sigma_p^2 d \|\boldsymbol{\mu}\|_2^2 \sigma_h^2 d_h \log \left( O(\frac{1}{\epsilon}) \right)}{N \exp(\langle \boldsymbol{q}_{n,i}^{(s)}, \boldsymbol{k}_+^{(s)} \rangle)},$$

similarly, we have

$$\langle \boldsymbol{q}_{n,i}^{(s+1)}, \boldsymbol{k}_-^{(s+1)} \rangle - \langle \boldsymbol{q}_{n,i}^{(s)}, \boldsymbol{k}_-^{(s)} \rangle \leq \frac{\eta C_{10} \sigma_p^2 d \|\boldsymbol{\mu}\|_2^2 \sigma_h^2 d_h \log \left( O(\frac{1}{\epsilon}) \right)}{N \exp(\langle \boldsymbol{q}_{n,i}^{(s)}, \boldsymbol{k}_-^{(s)} \rangle)}. \tag{276}$$

$$\langle \boldsymbol{q}_{n,i}^{(s+1)}, \boldsymbol{k}_{n,j}^{(s+1)} \rangle - \langle \boldsymbol{q}_{n,i}^{(s)}, \boldsymbol{k}_{n,j}^{(s)} \rangle$$

$$= \alpha_{n,i,+}^{(s)} \langle \boldsymbol{k}_+^{(s)}, \boldsymbol{k}_{n,j}^{(s)} \rangle + \alpha_{n,i,-}^{(s)} \langle \boldsymbol{k}_-^{(s)}, \boldsymbol{k}_{n,j}^{(s)} \rangle + \sum_{n'=1}^{N} \sum_{l=2}^{M} \alpha_{n,i,n',l}^{(s)} \langle \boldsymbol{k}_{n',l}^{(s)}, \boldsymbol{k}_{n,j}^{(s)} \rangle$$

$$+ \beta_{n,j,+}^{(s)} \langle \boldsymbol{q}_+^{(s)}, \boldsymbol{q}_{n,i}^{(s)} \rangle + \beta_{n,j,-}^{(s)} \langle \boldsymbol{q}_-^{(s)}, \boldsymbol{q}_{n,i}^{(s)} \rangle + \sum_{n'=1}^{N} \sum_{l=2}^{M} \beta_{n,j,n',l}^{(s)} \langle \boldsymbol{q}_{n',l}^{(s)}, \boldsymbol{q}_{n,i}^{(s)} \rangle$$

$$+ \left( \alpha_{n,i,+}^{(s)} \boldsymbol{k}_+^{(s)} + \alpha_{n,i,-}^{(s)} \boldsymbol{k}_-^{(s)} + \sum_{n'=1}^{N} \sum_{l=2}^{M} \alpha_{n,i,n',l}^{(s)} \boldsymbol{k}_{n',l}^{(s)} \right)$$

$$\cdot \left( \beta_{n,j,+}^{(s)} \boldsymbol{q}_+^{(s)\top} + \beta_{n,j,-}^{(s)} \boldsymbol{q}_-^{(s)\top} + \sum_{n'=1}^{N} \sum_{l=2}^{M} \beta_{n,j,n',l}^{(s)} \boldsymbol{q}_{n',l}^{(s)\top} \right) \tag{277}$$

$$= \alpha_{n,i,n,j}^{(s)} \|\boldsymbol{k}_{n,j}^{(s)}\|_2^2 + \beta_{n,j,n,i}^{(s)} \|\boldsymbol{q}_{n,i}^{(s)}\|_2^2$$

$$+ \{lower\ order\ term\}$$

$$\geq -\frac{2\eta C_9 \sigma_p^2 d \log \left( O(\frac{1}{\epsilon}) \right)}{N} \exp(\langle \boldsymbol{q}_{n,i}^{(s)}, \boldsymbol{k}_{n,j}^{(s)} \rangle) \cdot \Theta \left( \sigma_p^2 \sigma_h^2 d d_h \right)$$

$$+ \{lower\ order\ term\}$$

$$\geq -\frac{\eta C_{10} \sigma_p^4 d^2 \sigma_h^2 d_h \log \left( O(\frac{1}{\epsilon}) \right)}{N} \cdot \exp(\langle \boldsymbol{q}_{n,i}^{(s)}, \boldsymbol{k}_{n,j}^{(s)} \rangle).$$

## F.10 Bounds for the Sum of $\alpha$ and $\beta$

Assume that the propositions $\mathcal{F}(T_2), \ldots, \mathcal{F}(s), \mathcal{H}(T_2), \ldots, \mathcal{H}(s-1)$ hold ($s \in [T_1, t]$), we have

$$|V_\pm^{(s)}| \leq 2\log\left(O(\tfrac{1}{\epsilon})\right), \tag{278}$$

$$|V_{n,i}^{(s)}| = O(1), \tag{279}$$

$$\Lambda_{n,\pm,j}^{(s)} \geq \Lambda_{n,\pm,j}^{(T_2)} \geq \log\left(\exp(\Lambda_{n,\pm,j}^{(T_1)}) + \Theta\left(\frac{d_h^{\frac{1}{2}}}{N\left(\log(6N^2M^2/\delta)\right)^3}\right)\right), \tag{280}$$

$$\Lambda_{n,i,\pm,j}^{(s)} \geq \Lambda_{n,i,\pm,j}^{(T_2)} \geq \log\left(\exp(\Lambda_{n,i,\pm,j}^{(T_1)}) + \Theta\left(\frac{\sigma_p^2 d d_h^{\frac{1}{2}}}{N\|\boldsymbol{\mu}\|_2^2\left(\log(6N^2M^2/\delta)\right)^3}\right)\right) \tag{281}$$

for $i, j \in [M]\backslash\{1\}, n \in [N], s \in [T_2, t]$. Similar to (64) and (65), we have

$$\frac{\exp(\langle \boldsymbol{q}_\pm^{(s)}, \boldsymbol{k}_{n,j}^{(s)}\rangle)}{\exp(\langle \boldsymbol{q}_\pm^{(s)}, \boldsymbol{k}_\pm^{(s)}\rangle) + \sum_{j'=2}^{M} \exp(\langle \boldsymbol{q}_\pm^{(s)}, \boldsymbol{k}_{n,j'}^{(s)}\rangle)} = O\left(\frac{N\left(\log(6N^2M^2/\delta)\right)^3}{d_h^{\frac{1}{2}}}\right) \tag{282}$$

$$\frac{\exp(\langle \boldsymbol{q}_{n,i}^{(s)}, \boldsymbol{k}_{n,j}^{(s)}\rangle)}{\exp(\langle \boldsymbol{q}_{n,i}^{(s)}, \boldsymbol{k}_+^{(s)}\rangle) + \sum_{j'=2}^{M} \exp(\langle \boldsymbol{q}_{n,i}^{(s)}, \boldsymbol{k}_{n,j'}^{(s)}\rangle)} = O\left(\frac{N\|\boldsymbol{\mu}\|_2^2\left(\log(6N^2M^2/\delta)\right)^3}{\sigma_p^2 d d_h^{\frac{1}{2}}}\right) \tag{283}$$

Plugging (282),(283) into the expressions of $\alpha$, $\beta$ and letting $O\left(\log\left(O(\tfrac{1}{\epsilon})\right)\right)$ be the upper bound for $|V_\pm^{(s)}|, |V_{n,i}^{(s)}|$ we have

$$|\alpha_{+,+}^{(s)}| = \left| \frac{\eta}{NM} \sum_{n\in S_+} -\ell_n'^{(s)}\|\boldsymbol{\mu}\|_2^2 \right.$$

$$\cdot \left(V_+^{(s)}\left(\frac{\exp(\langle \boldsymbol{q}_+^{(s)}, \boldsymbol{k}_+^{(s)}\rangle)}{\exp(\langle \boldsymbol{q}_+^{(s)}, \boldsymbol{k}_+^{(s)}\rangle) + \sum_{j=2}^{M} \exp(\langle \boldsymbol{q}_+^{(s)}, \boldsymbol{k}_{n,j}^{(s)}\rangle)}\right.\right.$$

$$\left. - \left(\frac{\exp(\langle \boldsymbol{q}_+^{(s)}, \boldsymbol{k}_+^{(s)}\rangle)}{\exp(\langle \boldsymbol{q}_+^{(s)}, \boldsymbol{k}_+^{(s)}\rangle) + \sum_{j=2}^{M} \exp(\langle \boldsymbol{q}_+^{(s)}, \boldsymbol{k}_{n,j}^{(s)}\rangle)}\right)^2\right)$$

$$- \sum_{i=2}^{M} \left(V_{n,i}^{(s)} \cdot \frac{\exp(\langle \boldsymbol{q}_+^{(s)}, \boldsymbol{k}_+^{(s)}\rangle)}{\exp(\langle \boldsymbol{q}_+^{(s)}, \boldsymbol{k}_+^{(s)}\rangle) + \sum_{j=2}^{M} \exp(\langle \boldsymbol{q}_+^{(s)}, \boldsymbol{k}_{n,j}^{(s)}\rangle)}\right. \tag{284}$$

$$\left.\left.\left.\cdot \frac{\exp(\langle \boldsymbol{q}_+^{(s)}, \boldsymbol{k}_{n,i}^{(s)}\rangle)}{\exp(\langle \boldsymbol{q}_+^{(s)}, \boldsymbol{k}_+^{(s)}\rangle) + \sum_{j=2}^{M} \exp(\langle \boldsymbol{q}_+^{(s)}, \boldsymbol{k}_{n,j}^{(s)}\rangle)}\right)\right)\right|$$

$$\leq \frac{\eta\|\boldsymbol{\mu}\|_2^2}{NM} \cdot \frac{3N}{4} \cdot O\left(\log\left(O(\tfrac{1}{\epsilon})\right)\right) \cdot O\left(\frac{N\left(\log(6N^2M^2/\delta)\right)^3}{d_h^{\frac{1}{2}}}\right)$$

$$= O\left(\frac{\eta N\|\boldsymbol{\mu}\|_2^2\left(\log(6N^2M^2/\delta)\right)^3 \log\left(O(\tfrac{1}{\epsilon})\right)}{d_h^{\frac{1}{2}}}\right)$$

Similarly, we have

$$|\alpha_{-,-}^{(s)}|, |\beta_{+,+}^{(s)}|, |\beta_{-,-}^{(s)}| = O\Big(\frac{\eta N\|\boldsymbol{\mu}\|_2^2\big(\log(6N^2M^2/\delta)\big)^3\log\big(O(\frac{1}{\epsilon})\big)}{d_h^{\frac{1}{2}}}\Big), \tag{285}$$

$$|\alpha_{n,+,i}^{(s)}|, |\alpha_{n,-,i}^{(s)}| = O\Big(\frac{\eta\|\boldsymbol{\mu}\|_2^2\big(\log(6N^2M^2/\delta)\big)^3\log\big(O(\frac{1}{\epsilon})\big)}{d_h^{\frac{1}{2}}}\Big), \tag{286}$$

$$\begin{aligned}|\beta_{n,+,i}^{(s)}|, |\beta_{n,-,i}^{(s)}| &= O\Big(\frac{\eta\|\boldsymbol{\mu}\|_2^4\big(\log(6N^2M^2/\delta)\big)^3\log\big(O(\frac{1}{\epsilon})\big)}{\sigma_p^2 dd_h^{\frac{1}{2}}}\Big)\\&= O\Big(\frac{\eta\|\boldsymbol{\mu}\|_2^2\cdot\mathrm{SNR}^2\big(\log(6N^2M^2/\delta)\big)^3\log\big(O(\frac{1}{\epsilon})\big)}{d_h^{\frac{1}{2}}}\Big),\end{aligned} \tag{287}$$

for $i \in [M]\backslash\{1\}, n \in S_\pm$.

$$|\alpha_{n,i,+}^{(s)}|, |\alpha_{n,i,-}^{(s)}| = O\Big(\frac{\eta\|\boldsymbol{\mu}\|_2^2\big(\log(6N^2M^2/\delta)\big)^3\log\big(O(\frac{1}{\epsilon})\big)}{d_h^{\frac{1}{2}}}\Big), \tag{288}$$

for $i \in [M]\backslash\{1\}, n \in S_\pm$.

$$\begin{aligned}|\beta_{n,i,+}^{(s)}|, |\beta_{n,i,-}^{(s)}| &= O\Big(\frac{\eta\sigma_p^2 d\big(\log(6N^2M^2/\delta)\big)^3\log\big(O(\frac{1}{\epsilon})\big)}{d_h^{\frac{1}{2}}}\Big)\\&= O\Big(\frac{\eta N\|\boldsymbol{\mu}\|_2^2\big(\log(6N^2M^2/\delta)\big)^3\log\big(O(\frac{1}{\epsilon})\big)}{d_h^{\frac{1}{2}}}\Big),\end{aligned} \tag{289}$$

for $i \in [M]\backslash\{1\}, n \in S_\pm$, the last equality is by $N \cdot \mathrm{SNR}^2 \geq \Omega(1)$.

$$|\alpha_{n,i,n,j}^{(s)}|, |\beta_{n,j,n,i}^{(s)}| = O\Big(\frac{\eta\|\boldsymbol{\mu}\|_2^2\big(\log(6N^2M^2/\delta)\big)^3\log\big(O(\frac{1}{\epsilon})\big)}{d_h^{\frac{1}{2}}}\Big), \tag{290}$$

for $i, j \in [M]\backslash\{1\}, n \in [N]$.

$$|\alpha_{n,i,n',j}^{(s)}|, |\beta_{n,j,n',i}^{(s)}| = O\Big(\frac{\eta\|\boldsymbol{\mu}\|_2^2\big(\log(6N^2M^2/\delta)\big)^4\log\big(O(\frac{1}{\epsilon})\big)}{d^{\frac{1}{2}}d_h^{\frac{1}{2}}}\Big), \tag{291}$$

for $i, j \in [M]\backslash\{1\}, n, n' \in [N], n \neq n'$. Taking a summation we obtain that

$$\begin{aligned}\sum_{s=T_2}^{t}|\alpha_{+,+}^{(s)}| &= O\Big(\frac{1}{\eta\epsilon\|\boldsymbol{\mu}\|_2^2\|\boldsymbol{w}_O\|_2^2}\Big)\cdot O\Big(\frac{\eta\|\boldsymbol{\mu}\|_2^2 N\big(\log(6N^2M^2/\delta)\big)^3\log\big(O(\frac{1}{\epsilon})\big)}{d_h^{\frac{1}{2}}}\Big)\\&= O\Big(\frac{N\big(\log(6N^2M^2/\delta)\big)^3\log\big(O(\frac{1}{\epsilon})\big)}{\epsilon d_h^{\frac{1}{2}}}\Big),\end{aligned} \tag{292}$$

where the last equality is by $\|\boldsymbol{w}_O\| = \Theta(1)$. Similarly, we have

$$\sum_{s=T_2}^{t}|\beta_{n,+,i}^{(s)}|, \sum_{s=T_2}^{t}|\beta_{n,-,i}^{(s)}| = O\Big(\frac{\mathrm{SNR}^2\big(\log(6N^2M^2/\delta)\big)^3\log\big(O(\frac{1}{\epsilon})\big)}{\epsilon d_h^{\frac{1}{2}}}\Big), \tag{293}$$

for $i \in [M]\backslash\{1\}, n \in S_\pm$.

$$\sum_{s=T_2}^{t}|\alpha_{-,-}^{(s)}|, \sum_{s=T_2}^{t}|\beta_{+,+}^{(s)}|, \sum_{s=T_2}^{t}|\beta_{-,-}^{(s)}|, \sum_{s=T_2}^{t}|\beta_{n,+,i}^{(s)}|, \sum_{s=T_2}^{t}|\beta_{n,-,i}^{(s)}|, \sum_{s=T_2}^{t}|\beta_{n,i,+}^{(s)}|, \sum_{s=T_2}^{t}|\beta_{n,i,-}^{(s)}|$$

$$= O\Big(\frac{N\big(\log(6N^2M^2/\delta)\big)^3\log\big(O(\frac{1}{\epsilon})\big)}{\epsilon d_h^{\frac{1}{2}}}\Big), \tag{294}$$

for $i \in [M]\backslash\{1\}, n \in S_\pm$.

$$\sum_{s=T_2}^{t} |\alpha_{n,+,i}^{(s)}|, \sum_{s=T_2}^{t} |\alpha_{n,-,i}^{(s)}|, \sum_{s=T_2}^{t} |\alpha_{n,i,+}^{(s)}|, \sum_{s=T_2}^{t} |\alpha_{n,i,-}^{(s)}|, \sum_{s=T_2}^{t} |\alpha_{n,i,n,j}^{(s)}|, \sum_{s=T_2}^{t} |\beta_{n,j,n,i}^{(s)}|$$
$$= O\Big( \frac{\big( \log(6N^2 M^2/\delta) \big)^3 \log \big( O(\frac{1}{\epsilon}) \big)}{\epsilon d_h^{\frac{1}{2}}} \Big), \tag{295}$$

for $i, j \in [M]\backslash\{1\}, n \in S_\pm$.

$$\sum_{s=T_2}^{t} |\alpha_{n,i,n',j}^{(s)}|, \sum_{s=T_2}^{t} |\beta_{n,j,n',i}^{(s)}| = O\Big( \frac{\big( \log(6N^2 M^2/\delta) \big)^4 \log \big( O(\frac{1}{\epsilon}) \big)}{\epsilon d^{\frac{1}{2}} d_h^{\frac{1}{2}}} \Big) \tag{296}$$

for $i, j \in [M]\backslash\{1\}, n, n' \in [N], n \neq n'$.

With these sums of $\alpha$ and $\beta$ above, we can easily prove **Claim** 7 and **Claim** 8.

### F.11  Proof of Claim 7

In this subsection, we assume that $\mathcal{I}(T_2), \ldots, \mathcal{I}(t)$ hold, and then proof that $\mathcal{G}(t+1)$ is true with the result of F.10.

$$\left| \|\boldsymbol{q}_+^{(t+1)}\|_2^2 - \|\boldsymbol{q}_+^{(t+1)}\|_2^2 \right| \leq \sum_{s=T_2}^{t} \left| \|\boldsymbol{q}_+^{(s+1)}\|_2^2 - \|\boldsymbol{q}_+^{(s)}\|_2^2 \right|$$

$$\leq \sum_{s=T_2}^{t} \left| 2\alpha_{+,+}^{(s)} \langle \boldsymbol{q}_+^{(s)}, \boldsymbol{k}_+^{(s)} \rangle + 2 \sum_{n \in S_+} \sum_{i=2}^{M} \alpha_{n,+,i}^{(s)} \langle \boldsymbol{q}_+^{(s)}, \boldsymbol{k}_{n,i}^{(s)} \rangle \right.$$

$$+ \Big( \alpha_{+,+}^{(s)} \boldsymbol{k}_+^{(s)} + \sum_{n \in S_+} \sum_{i=2}^{M} \alpha_{n,+,i}^{(s)} \boldsymbol{k}_{n,i}^{(s)} \Big)$$

$$\left. \cdot \Big( \alpha_{+,+}^{(s)} \boldsymbol{k}_+^{(s)\top} + \sum_{n \in S_+} \sum_{i=2}^{M} \alpha_{n,+,i}^{(s)} \boldsymbol{k}_{n,i}^{(s)\top} \Big) \right|$$

$$\leq 2 \sum_{s=T_2}^{t} |\alpha_{+,+}^{(s)}| |\langle \boldsymbol{q}_+^{(s)}, \boldsymbol{k}_+^{(s)} \rangle| + 2 \sum_{n \in S_+} \sum_{i=2}^{M} \sum_{s=T_2}^{t} |\alpha_{n,+,i}^{(s)}| |\langle \boldsymbol{q}_+^{(s)}, \boldsymbol{k}_{n,i}^{(s)} \rangle|$$

$$+ \{lower\ order\ term\}$$

$$= O\Big( \frac{N \big( \log(6N^2 M^2/\delta) \big)^3 \log \big( O(\frac{1}{\epsilon}) \big)}{\epsilon d_h^{\frac{1}{2}}} \Big) \cdot \log(\epsilon^{-1} d_h^{\frac{1}{2}})$$

$$+ N \cdot M \cdot O\Big( \frac{\big( \log(6N^2 M^2/\delta) \big)^3 \log \big( O(\frac{1}{\epsilon}) \big)}{\epsilon d_h^{\frac{1}{2}}} \Big) \cdot \log(\epsilon^{-1} d_h^{\frac{1}{2}})$$

$$= O\Big( \frac{N \big( \log(6N^2 M^2/\delta) \big)^3 \log \big( O(\frac{1}{\epsilon}) \big) \log(\epsilon^{-1} d_h^{\frac{1}{2}})}{\epsilon d_h^{\frac{1}{2}}} \Big)$$

where the first inequality is by triangle inequality, the second inequality is by 199, the third inequality is by $t \leq T_3$. Since $\sigma_h^2 \geq \big( \max\{\sigma_p^2 d, \|\mu\|_2^2\} \big)^{-1} \cdot d_h^{-\frac{1}{2}} (\log(6N^2 M^2/\delta))^{-2}$ and $d_h = \widetilde{\Omega}\Big( \max\{\mathrm{SNR}^4, \mathrm{SNR}^{-4}\} N^2 \epsilon^{-2} \Big)$, we have $\frac{N \big( \log(6N^2 M^2/\delta) \big)^3 \log \big( O(\frac{1}{\epsilon}) \big) \log(\epsilon^{-1} d_h^{\frac{1}{2}})}{\epsilon d_h^{\frac{1}{2}}} =$

$o(\|\boldsymbol{\mu}\|_2^2\sigma_h^2 d_h)$, so $\|\boldsymbol{q}_+^{(t+1)}\|_2^2 = \|\boldsymbol{q}_+^{(T_2)}\|_2^2 + o(\|\boldsymbol{\mu}\|_2^2\sigma_h^2 d_h) = \Theta(\|\boldsymbol{\mu}\|_2^2\sigma_h^2 d_h)$. Similarly, we have

$$\left|\|\boldsymbol{q}_-^{(t+1)}\|_2^2 - \|\boldsymbol{q}_-^{(t+1)}\|_2^2\right| = O\Big(\frac{N\big(\log(6N^2M^2/\delta)\big)^3 \log\big(O(\frac{1}{\epsilon})\big)\log(\epsilon^{-1}d_h^{\frac{1}{2}})}{\epsilon d_h^{\frac{1}{2}}}\Big) = o(\|\boldsymbol{\mu}\|_2^2\sigma_h^2 d_h),$$

$$\left|\|\boldsymbol{k}_\pm^{(t+1)}\|_2^2 - \|\boldsymbol{k}_\pm^{(t+1)}\|_2^2\right| = O\Big(\frac{(1+SNR^2)N\big(\log(6N^2M^2/\delta)\big)^3 \log\big(O(\frac{1}{\epsilon})\big)\log(\epsilon^{-1}d_h^{\frac{1}{2}})}{\epsilon d_h^{\frac{1}{2}}}\Big) = o(\|\boldsymbol{\mu}\|_2^2\sigma_h^2 d_h),$$

$$\left|\|\boldsymbol{q}_{n,i}^{(t+1)}\|_2^2 - \|\boldsymbol{q}_{n,i}^{(t+1)}\|_2^2\right| = O\Big(\frac{\big(\log(6N^2M^2/\delta)\big)^3 \log\big(O(\frac{1}{\epsilon})\big)\log(\epsilon^{-1}d_h^{\frac{1}{2}})}{\epsilon d_h^{\frac{1}{2}}}\Big) = o(\sigma_p^2\sigma_h^2 dd_h),$$

$$\left|\|\boldsymbol{k}_{n,i}^{(t+1)}\|_2^2 - \|\boldsymbol{k}_{n,i}^{(t+1)}\|_2^2\right| = O\Big(\frac{N\big(\log(6N^2M^2/\delta)\big)^3 \log\big(O(\frac{1}{\epsilon})\big)\log(\epsilon^{-1}d_h^{\frac{1}{2}})}{\epsilon d_h^{\frac{1}{2}}}\Big) = o(\sigma_p^2\sigma_h^2 dd_h),$$

so we have

$$\|\boldsymbol{q}_\pm^{(t+1)}\|_2^2, \|\boldsymbol{k}_\pm^{(t+1)}\|_2^2 = \Theta(\|\boldsymbol{\mu}\|_2^2\sigma_h^2 d_h),$$

$$\|\boldsymbol{q}_{n,i}^{(t+1)}\|_2^2, \|\boldsymbol{k}_{n,i}^{(t+1)}\|_2^2 = \Theta(\sigma_p^2\sigma_h^2 dd_h)$$

for $i \in [M]\backslash\{1\}, n \in [N]$.

$$|\langle\boldsymbol{q}_+^{(t+1)}, \boldsymbol{q}_-^{(t+1)}\rangle| \le |\langle\boldsymbol{q}_+^{(T_2)}, \boldsymbol{q}_-^{(T_2)}\rangle| + \sum_{s=T_2}^t \left|\langle\boldsymbol{q}_+^{(s+1)}, \boldsymbol{q}_-^{(s+1)}\rangle - \langle\boldsymbol{q}_+^{(s)}, \boldsymbol{q}_-^{(s)}\rangle\right|$$

$$\le |\langle\boldsymbol{q}_+^{(T_2)}, \boldsymbol{q}_-^{(T_2)}\rangle|$$

$$+ \sum_{s=T_2}^t \left|\alpha_{+,+}^{(s)}\langle\boldsymbol{q}_-^{(s)}, \boldsymbol{k}_+^{(s)}\rangle + \sum_{n\in S_+}\sum_{i=2}^M \alpha_{n,+,i}^{(s)}\langle\boldsymbol{q}_-^{(s)}, \boldsymbol{k}_{n,i}^{(s)}\rangle\right.$$

$$+ \alpha_{-,-}^{(s)}\langle\boldsymbol{q}_+^{(s)}, \boldsymbol{k}_-^{(s)}\rangle + \sum_{n\in S_-}\sum_{i=2}^M \alpha_{n,-,i}^{(s)}\langle\boldsymbol{q}_+^{(s)}, \boldsymbol{k}_{n,i}^{(s)}\rangle$$

$$+ \Big(\alpha_{+,+}^{(s)}\boldsymbol{k}_+^{(s)} + \sum_{n\in S_+}\sum_{i=2}^M \alpha_{n,+,i}^{(s)}\boldsymbol{k}_{n,i}^{(s)}\Big)$$

$$\left.\cdot \Big(\alpha_{-,-}^{(s)}\boldsymbol{k}_-^{(s)\top} + \sum_{n\in S_-}\sum_{i=2}^M \alpha_{n,-,i}^{(s)}\boldsymbol{k}_{n,i}^{(s)\top}\Big)\right|$$

$$\le |\langle\boldsymbol{q}_+^{(T_2)}, \boldsymbol{q}_-^{(T_2)}\rangle|$$

$$+ \sum_{s=T_2}^t |\alpha_{+,+}^{(s)}||\langle\boldsymbol{q}_-^{(s)}, \boldsymbol{k}_+^{(s)}\rangle| + \sum_{n\in S_+}\sum_{i=2}^M\sum_{s=T_2}^t |\alpha_{n,+,i}^{(s)}||\langle\boldsymbol{q}_-^{(s)}, \boldsymbol{k}_{n,i}^{(s)}\rangle|$$

$$+ \sum_{s=T_2}^t |\alpha_{-,-}^{(s)}||\langle\boldsymbol{q}_+^{(s)}, \boldsymbol{k}_-^{(s)}\rangle| + \sum_{n\in S_-}\sum_{i=2}^M\sum_{s=T_2}^t |\alpha_{n,-,i}^{(s)}||\langle\boldsymbol{q}_+^{(s)}, \boldsymbol{k}_{n,i}^{(s)}\rangle|$$

$$+ \{lower\ order\ term\}$$

$$\le |\langle\boldsymbol{q}_+^{(T_2)}, \boldsymbol{q}_-^{(T_2)}\rangle|$$

$$+ O\Big(\frac{N\big(\log(6N^2M^2/\delta)\big)^3 \log\big(O(\frac{1}{\epsilon})\big)}{\epsilon d_h^{\frac{1}{2}}}\Big) \cdot o(1)$$

$$+ N \cdot M \cdot O\Big(\frac{\big(\log(6N^2M^2/\delta)\big)^3 \log\big(O(\frac{1}{\epsilon})\big)}{\epsilon d_h^{\frac{1}{2}}}\Big) \cdot \log(\epsilon^{-1}d_h^{\frac{1}{2}})$$

$$= |\langle \boldsymbol{q}_+^{(T_2)}, \boldsymbol{q}_-^{(T_2)}\rangle| + O\Big(\frac{N\big(\log(6N^2M^2/\delta)\big)^3 \log\big(O(\frac{1}{\epsilon})\big)\log(\epsilon^{-1}d_h^{\frac{1}{2}})}{\epsilon d_h^{\frac{1}{2}}}\Big)$$

$$= o(1),$$

where the first inequality is triangle inequality, the second inequality is by (198), the last equality is by $d_h = \widetilde{\Omega}\Big(\max\{\text{SNR}^4, \text{SNR}^{-4}\}N^2\epsilon^{-2}\Big)$.

$$|\langle \boldsymbol{q}_+^{(t+1)}, \boldsymbol{q}_{n,i}^{(t+1)}\rangle| \le |\langle \boldsymbol{q}_+^{(T_2)}, \boldsymbol{q}_{n,i}^{(T_2)}\rangle| + \sum_{s=T_2}^{t} \Big|\langle \boldsymbol{q}_+^{(s+1)}, \boldsymbol{q}_{n,i}^{(s+1)}\rangle - \langle \boldsymbol{q}_+^{(s)}, \boldsymbol{q}_{n,i}^{(s)}\rangle\Big|$$

$$\le |\langle \boldsymbol{q}_+^{(T_2)}, \boldsymbol{q}_{n,i}^{(T_2)}\rangle|$$

$$+ \sum_{s=T_2}^{t} \Big| \alpha_{+,+}^{(s)}\langle \boldsymbol{q}_{n,i}^{(s)}, \boldsymbol{k}_+^{(s)}\rangle + \sum_{n'\in S_+}\sum_{l=2}^{M} \alpha_{n',+,l}^{(s)}\langle \boldsymbol{q}_{n,i}^{(s)}, \boldsymbol{k}_{n',l}^{(s)}\rangle$$

$$+ \alpha_{n,i,+}^{(s)}\langle \boldsymbol{q}_+^{(s)}, \boldsymbol{k}_+^{(s)}\rangle + \alpha_{n,i,-}^{(s)}\langle \boldsymbol{q}_+^{(s)}, \boldsymbol{k}_-^{(s)}\rangle + \sum_{n'=1}^{N}\sum_{l=2}^{M} \alpha_{n,i,n',l}^{(s)}\langle \boldsymbol{q}_+^{(s)}, \boldsymbol{k}_{n',l}^{(s)}\rangle$$

$$+ \Big(\alpha_{+,+}^{(s)}\boldsymbol{k}_+^{(s)} + \sum_{n\in S_+}\sum_{i=2}^{M}\alpha_{n,+,i}^{(s)}\boldsymbol{k}_{n,i}^{(s)}\Big)$$

$$\cdot \Big(\alpha_{n,i,+}^{(s)}\boldsymbol{k}_+^{(s)\top} + \alpha_{n,i,-}^{(s)}\boldsymbol{k}_-^{(s)\top} + \sum_{n'=1}^{N}\sum_{l=2}^{M}\alpha_{n,i,n',l}^{(s)}\boldsymbol{k}_{n',l}^{(s)\top}\Big)\Big|$$

$$\le |\langle \boldsymbol{q}_+^{(T_2)}, \boldsymbol{q}_{n,i}^{(T_2)}\rangle|$$

$$+ \sum_{s=T_2}^{t} |\alpha_{+,+}^{(s)}||\langle \boldsymbol{q}_{n,i}^{(s)}, \boldsymbol{k}_+^{(s)}\rangle| + \sum_{l=2}^{M}\sum_{s=T_2}^{t} |\alpha_{n,+,l}^{(s)}||\langle \boldsymbol{q}_{n,i}^{(s)}, \boldsymbol{k}_{n,l}^{(s)}\rangle| + \sum_{n'\in S_+\wedge n'\neq n}\sum_{l=2}^{M}\sum_{s=T_2}^{t} |\alpha_{n',+,l}^{(s)}||\langle \boldsymbol{q}_{n,i}^{(s)}, \boldsymbol{k}_{n',l}^{(s)}\rangle|$$

$$+ \sum_{s=T_2}^{t} |\alpha_{n,i,+}^{(s)}||\langle \boldsymbol{q}_+^{(s)}, \boldsymbol{k}_+^{(s)}\rangle| + \sum_{s=T_2}^{t} |\alpha_{n,i,-}^{(s)}||\langle \boldsymbol{q}_+^{(s)}, \boldsymbol{k}_-^{(s)}\rangle| + \sum_{l=2}^{M}\sum_{s=T_2}^{t} |\alpha_{n,i,n,l}^{(s)}||\langle \boldsymbol{q}_+^{(s)}, \boldsymbol{k}_{n,l}^{(s)}\rangle|$$

$$+ \sum_{n'\neq n}\sum_{l=2}^{M}\sum_{s=T_2}^{t} |\alpha_{n,i,n',l}^{(s)}||\langle \boldsymbol{q}_+^{(s)}, \boldsymbol{k}_{n',l}^{(s)}\rangle|$$

$$+ \{lower\ order\ term\}$$

$$\le |\langle \boldsymbol{q}_+^{(T_2)}, \boldsymbol{q}_{n,i}^{(T_2)}\rangle|$$

$$+ O\Big(\frac{N\big(\log(6N^2M^2/\delta)\big)^3 \log\big(O(\frac{1}{\epsilon})\big)}{\epsilon d_h^{\frac{1}{2}}}\Big)\cdot \log(\epsilon^{-1}d_h^{\frac{1}{2}}) + M\cdot O\Big(\frac{\big(\log(6N^2M^2/\delta)\big)^3 \log\big(O(\frac{1}{\epsilon})\big)}{\epsilon d_h^{\frac{1}{2}}}\Big)\cdot \log(\epsilon^{-1}d_h^{\frac{1}{2}})$$

$$+ N\cdot M\cdot O\Big(\frac{\big(\log(6N^2M^2/\delta)\big)^3 \log\big(O(\frac{1}{\epsilon})\big)}{\epsilon d_h^{\frac{1}{2}}}\Big)\cdot o(1)$$

$$+ M\cdot O\Big(\frac{\big(\log(6N^2M^2/\delta)\big)^3 \log\big(O(\frac{1}{\epsilon})\big)}{\epsilon d_h^{\frac{1}{2}}}\Big)\cdot \log(\epsilon^{-1}d_h^{\frac{1}{2}})$$

$$+ N\cdot M\cdot O\Big(\frac{\big(\log(6N^2M^2/\delta)\big)^4 \log\big(O(\frac{1}{\epsilon})\big)}{\epsilon d^{\frac{1}{2}} d_h^{\frac{1}{2}}}\Big)\cdot \log(\epsilon^{-1}d_h^{\frac{1}{2}})$$

$$= |\langle \boldsymbol{q}_+^{(T_2)}, \boldsymbol{q}_{n,i}^{(T_2)}\rangle| + O\Big(\frac{N\big(\log(6N^2M^2/\delta)\big)^3 \log\big(O(\frac{1}{\epsilon})\big)\log(\epsilon^{-1}d_h^{\frac{1}{2}})}{\epsilon d_h^{\frac{1}{2}}}\Big)$$

$$+ O\Big(\frac{N\big(\log(6N^2M^2/\delta)\big)^4 \log\big(O(\frac{1}{\epsilon})\big)\log(\epsilon^{-1}d_h^{\frac{1}{2}})}{\epsilon d^{\frac{1}{2}} d_h^{\frac{1}{2}}}\Big)$$

$$= o(1),$$

where the first inequality is triangle inequality, the second inequality is by (202), the last equality is by $d_h = \widetilde{\Omega}\Big(\max\{\mathrm{SNR}^4, \mathrm{SNR}^{-4}\}N^2\epsilon^{-2}\Big)$ and $d = \widetilde{\Omega}\Big(\epsilon^{-2}N^2 d_h\Big)$. Similarly, we have $|\langle \boldsymbol{q}_-^{(t+1)}, \boldsymbol{q}_{n,i}^{(t+1)}\rangle| = o(1)$.

$$|\langle \boldsymbol{q}_{n,i}^{(t+1)}, \boldsymbol{q}_{n,j}^{(t+1)}\rangle| \le |\langle \boldsymbol{q}_{n,i}^{(T_2)}, \boldsymbol{q}_{n,j}^{(T_2)}\rangle| + \sum_{s=T_2}^{t} \left| \langle \boldsymbol{q}_{n,i}^{(s+1)}, \boldsymbol{q}_{n,j}^{(s+1)}\rangle - \langle \boldsymbol{q}_{n,i}^{(s)}, \boldsymbol{q}_{n,j}^{(s)}\rangle \right|$$

$$\le |\langle \boldsymbol{q}_{n,i}^{(T_2)}, \boldsymbol{q}_{n,j}^{(T_2)}\rangle|$$

$$+ \sum_{s=T_2}^{t} \Big| \alpha_{n,i,+}^{(s)}\langle \boldsymbol{q}_{n,j}^{(s)}, \boldsymbol{k}_+^{(s)}\rangle + \alpha_{n,i,-}^{(s)}\langle \boldsymbol{q}_{n,j}^{(s)}, \boldsymbol{k}_-^{(s)}\rangle + \sum_{n'=1}^{N}\sum_{l=2}^{M} \alpha_{n,i,n',l}^{(s)}\langle \boldsymbol{q}_{n,j}^{(s)}, \boldsymbol{k}_{n',l}^{(s)}\rangle$$

$$+ \alpha_{n,j,+}^{(s)}\langle \boldsymbol{q}_{n,i}^{(s)}, \boldsymbol{k}_+^{(s)}\rangle + \alpha_{n,j,-}^{(s)}\langle \boldsymbol{q}_{n,i}^{(s)}, \boldsymbol{k}_-^{(s)}\rangle + \sum_{n'=1}^{N}\sum_{l=2}^{M} \alpha_{n,j,n',l}^{(s)}\langle \boldsymbol{q}_{n,i}^{(s)}, \boldsymbol{k}_{n',l}^{(s)}\rangle$$

$$+ \Big( \alpha_{n,i,+}^{(s)}\boldsymbol{k}_+^{(s)} + \alpha_{n,i,-}^{(s)}\boldsymbol{k}_-^{(s)} + \sum_{n'=1}^{N}\sum_{l=2}^{M} \alpha_{n,i,n',l}^{(s)}\boldsymbol{k}_{n',l}^{(s)} \Big)$$

$$\cdot \Big( \alpha_{n,j,+}^{(s)}\boldsymbol{k}_+^{(s)\top} + \alpha_{n,j,-}^{(s)}\boldsymbol{k}_-^{(s)\top} + \sum_{n'=1}^{N}\sum_{l=2}^{M} \alpha_{n,j,n',l}^{(s)}\boldsymbol{k}_{n',l}^{(s)\top} \Big) \Big|$$

$$\le |\langle \boldsymbol{q}_{n,i}^{(T_2)}, \boldsymbol{q}_{n,j}^{(T_2)}\rangle|$$

$$+ \sum_{s=T_2}^{t} |\alpha_{n,i,+}^{(s)}||\langle \boldsymbol{q}_{n,j}^{(s)}, \boldsymbol{k}_+^{(s)}\rangle| + \sum_{s=T_2}^{t} |\alpha_{n,i,-}^{(s)}||\langle \boldsymbol{q}_{n,j}^{(s)}, \boldsymbol{k}_-^{(s)}\rangle| + \sum_{l=2}^{M}\sum_{s=T_2}^{t} |\alpha_{n,i,n,l}^{(s)}||\langle \boldsymbol{q}_{n,j}^{(s)}, \boldsymbol{k}_{n,l}^{(s)}\rangle|$$

$$+ \sum_{n'\ne n}\sum_{l=2}^{M}\sum_{s=T_2}^{t} |\alpha_{n,i,n',l}^{(s)}||\langle \boldsymbol{q}_{n,j}^{(s)}, \boldsymbol{k}_{n',l}^{(s)}\rangle| + \sum_{s=T_2}^{t} |\alpha_{n,j,+}^{(s)}||\langle \boldsymbol{q}_{n,i}^{(s)}, \boldsymbol{k}_+^{(s)}\rangle| + \sum_{s=T_2}^{t} |\alpha_{n,j,-}^{(s)}||\langle \boldsymbol{q}_{n,i}^{(s)}, \boldsymbol{k}_-^{(s)}\rangle|$$

$$+ \sum_{l=2}^{M}\sum_{s=T_2}^{t} |\alpha_{n,j,n,l}^{(s)}||\langle \boldsymbol{q}_{n,i}^{(s)}, \boldsymbol{k}_{n,l}^{(s)}\rangle| + \sum_{n'\ne n}\sum_{l=2}^{M}\sum_{s=T_2}^{t} |\alpha_{n,j,n',l}^{(s)}||\langle \boldsymbol{q}_{n,i}^{(s)}, \boldsymbol{k}_{n',l}^{(s)}\rangle|$$

$$+ \{lower\ order\ term\}$$

$$\le |\langle \boldsymbol{q}_{n,i}^{(T_2)}, \boldsymbol{q}_{n,j}^{(T_2)}\rangle|$$

$$+ O\Big( \frac{\big(\log(6N^2 M^2/\delta)\big)^3 \log\big(O(\frac{1}{\epsilon})\big)}{\epsilon d_h^{\frac{1}{2}}} \Big) \cdot \log(\epsilon^{-1} d_h^{\frac{1}{2}})$$

$$+ M \cdot O\Big( \frac{\big(\log(6N^2 M^2/\delta)\big)^3 \log\big(O(\frac{1}{\epsilon})\big)}{\epsilon d_h^{\frac{1}{2}}} \Big) \cdot \log(\epsilon^{-1} d_h^{\frac{1}{2}})$$

$$+ N \cdot M \cdot O\Big( \frac{\big(\log(6N^2 M^2/\delta)\big)^4 \log\big(O(\frac{1}{\epsilon})\big)}{\epsilon d^{\frac{1}{2}} d_h^{\frac{1}{2}}} \Big) \cdot o(1)$$

$$= |\langle \boldsymbol{q}_{n,i}^{(T_2)}, \boldsymbol{q}_{n,j}^{(T_2)}\rangle| + O\Big( \frac{\big(\log(6N^2 M^2/\delta)\big)^3 \log\big(O(\frac{1}{\epsilon})\big) \log(\epsilon^{-1} d_h^{\frac{1}{2}})}{\epsilon d_h^{\frac{1}{2}}} \Big)$$

$$+ o\Big( \frac{N\big(\log(6N^2 M^2/\delta)\big)^4 \log\big(O(\frac{1}{\epsilon})\big)}{\epsilon d^{\frac{1}{2}} d_h^{\frac{1}{2}}} \Big)$$

$$= o(1)$$

for $i, j \in [M]\backslash\{1\}, i \neq j, n \in [N]$. The first inequality is triangle inequality, the second inequality is by (204), the last equality is by $d_h = \widetilde{\Omega}\Big( \max\{\mathrm{SNR}^4, \mathrm{SNR}^{-4}\}N^2\epsilon^{-2}\Big)$ and $d = \widetilde{\Omega}\Big(\epsilon^{-2}N^2 d_h\Big)$.

$$|\langle \boldsymbol{q}_{n,i}^{(t+1)}, \boldsymbol{q}_{\overline{n},j}^{(t+1)}\rangle| \leq |\langle \boldsymbol{q}_{n,i}^{(T_2)}, \boldsymbol{q}_{\overline{n},j}^{(T_2)}\rangle| + \sum_{s=T_2}^{t} \left| \langle \boldsymbol{q}_{n,i}^{(s+1)}, \boldsymbol{q}_{\overline{n},j}^{(s+1)}\rangle - \langle \boldsymbol{q}_{n,i}^{(s)}, \boldsymbol{q}_{\overline{n},j}^{(s)}\rangle \right|$$

$$\leq |\langle \boldsymbol{q}_{n,i}^{(T_2)}, \boldsymbol{q}_{\overline{n},j}^{(T_2)}\rangle|$$

$$+ \sum_{s=T_2}^{t} \left| \alpha_{n,i,+}^{(s)} \langle \boldsymbol{q}_{\overline{n},j}^{(s)}, \boldsymbol{k}_{+}^{(s)}\rangle + \alpha_{n,i,-}^{(s)} \langle \boldsymbol{q}_{\overline{n},j}^{(s)}, \boldsymbol{k}_{-}^{(s)}\rangle + \sum_{n'=1}^{N}\sum_{l=2}^{M} \alpha_{n,i,n',l}^{(s)}\langle \boldsymbol{q}_{\overline{n},j}^{(s)}, \boldsymbol{k}_{n',l}^{(s)}\rangle \right.$$

$$+ \alpha_{\overline{n},j,+}^{(s)} \langle \boldsymbol{q}_{n,i}^{(s)}, \boldsymbol{k}_{+}^{(s)}\rangle + \alpha_{\overline{n},j,-}^{(s)} \langle \boldsymbol{q}_{n,i}^{(s)}, \boldsymbol{k}_{-}^{(s)}\rangle + \sum_{n'=1}^{N}\sum_{l=2}^{M} \alpha_{\overline{n},j,n',l}^{(s)}\langle \boldsymbol{q}_{n,i}^{(s)}, \boldsymbol{k}_{n',l}^{(s)}\rangle$$

$$+ \left( \alpha_{n,i,+}^{(s)} \boldsymbol{k}_{+}^{(s)} + \alpha_{n,i,-}^{(s)} \boldsymbol{k}_{-}^{(s)} + \sum_{n'=1}^{N}\sum_{l=2}^{M} \alpha_{n,i,n',l}^{(s)}\boldsymbol{k}_{n',l}^{(s)} \right)$$

$$\left. \cdot \left( \alpha_{\overline{n},j,+}^{(s)} \boldsymbol{k}_{+}^{(s)\top} + \alpha_{\overline{n},j,-}^{(s)} \boldsymbol{k}_{-}^{(s)\top} + \sum_{n'=1}^{N}\sum_{l=2}^{M} \alpha_{\overline{n},j,n',l}^{(s)}\boldsymbol{k}_{n',l}^{(s)\top} \right) \right|$$

$$\leq |\langle \boldsymbol{q}_{n,i}^{(T_2)}, \boldsymbol{q}_{\overline{n},j}^{(T_2)}\rangle|$$

$$+ \sum_{s=T_2}^{t} |\alpha_{n,i,+}^{(s)}||\langle \boldsymbol{q}_{\overline{n},j}^{(s)}, \boldsymbol{k}_{+}^{(s)}\rangle| + \sum_{s=T_2}^{t} |\alpha_{n,i,-}^{(s)}||\langle \boldsymbol{q}_{\overline{n},j}^{(s)}, \boldsymbol{k}_{-}^{(s)}\rangle| + \sum_{l=2}^{M}\sum_{s=T_2}^{t} |\alpha_{n,i,\overline{n},l}^{(s)}||\langle \boldsymbol{q}_{\overline{n},j}^{(s)}, \boldsymbol{k}_{\overline{n},l}^{(s)}\rangle|$$

$$+ \sum_{l=2}^{M}\sum_{s=T_2}^{t} |\alpha_{n,i,n,l}^{(s)}||\langle \boldsymbol{q}_{\overline{n},j}^{(s)}, \boldsymbol{k}_{n,l}^{(s)}\rangle| + \sum_{n'\neq n \wedge n'\neq \overline{n}}\sum_{l=2}^{M}\sum_{s=T_2}^{t} |\alpha_{n,i,n',l}^{(s)}||\langle \boldsymbol{q}_{\overline{n},j}^{(s)}, \boldsymbol{k}_{n',l}^{(s)}\rangle|$$

$$+ \sum_{s=T_2}^{t} |\alpha_{\overline{n},j,+}^{(s)}||\langle \boldsymbol{q}_{n,i}^{(s)}, \boldsymbol{k}_{+}^{(s)}\rangle| + \sum_{s=T_2}^{t} |\alpha_{\overline{n},j,-}^{(s)}||\langle \boldsymbol{q}_{n,i}^{(s)}, \boldsymbol{k}_{-}^{(s)}\rangle| + \sum_{l=2}^{M}\sum_{s=T_2}^{t} |\alpha_{\overline{n},j,n,l}^{(s)}||\langle \boldsymbol{q}_{n,i}^{(s)}, \boldsymbol{k}_{n,l}^{(s)}\rangle|$$

$$+ \sum_{l=2}^{M}\sum_{s=T_2}^{t} |\alpha_{\overline{n},j,\overline{n},l}^{(s)}||\langle \boldsymbol{q}_{n,i}^{(s)}, \boldsymbol{k}_{\overline{n},l}^{(s)}\rangle| + \sum_{n'\neq n \wedge n'\neq \overline{n}}\sum_{l=2}^{M}\sum_{s=T_2}^{t} |\alpha_{\overline{n},j,n',l}^{(s)}||\langle \boldsymbol{q}_{n,i}^{(s)}, \boldsymbol{k}_{n',l}^{(s)}\rangle|$$

$$+ \{lower\ order\ term\}$$

$$= |\langle \boldsymbol{q}_{n,i}^{(T_2)}, \boldsymbol{q}_{\overline{n},j}^{(T_2)}\rangle|$$

$$+ O\Big( \frac{\big(\log(6N^2M^2/\delta)\big)^3 \log\big(O(\frac{1}{\epsilon})\big)}{\epsilon d_h^{\frac{1}{2}}} \Big) \cdot \log(\epsilon^{-1}d_h^{\frac{1}{2}})$$

$$+ M \cdot O\Big( \frac{\big(\log(6N^2M^2/\delta)\big)^4 \log\big(O(\frac{1}{\epsilon})\big)}{\epsilon d^{\frac{1}{2}} d_h^{\frac{1}{2}}} \Big) \cdot \log(\epsilon^{-1}d_h^{\frac{1}{2}})$$

$$+ M \cdot O\Big( \frac{\big(\log(6N^2M^2/\delta)\big)^3 \log\big(O(\frac{1}{\epsilon})\big)}{\epsilon d_h^{\frac{1}{2}}} \Big) \cdot o(1) + N \cdot M \cdot O\Big( \frac{\big(\log(6N^2M^2/\delta)\big)^4 \log\big(O(\frac{1}{\epsilon})\big)}{\epsilon d^{\frac{1}{2}} d_h^{\frac{1}{2}}} \Big) \cdot o(1)$$

$$= |\langle \boldsymbol{q}_{n,i}^{(T_2)}, \boldsymbol{q}_{\overline{n},j}^{(T_2)}\rangle| + O\Big( \frac{\big(\log(6N^2M^2/\delta)\big)^3 \log\big(O(\frac{1}{\epsilon})\big) \log(\epsilon^{-1}d_h^{\frac{1}{2}})}{\epsilon d_h^{\frac{1}{2}}} \Big)$$

$$+ o\Big( \frac{N\big(\log(6N^2M^2/\delta)\big)^4 \log\big(O(\frac{1}{\epsilon})\big)}{\epsilon d^{\frac{1}{2}} d_h^{\frac{1}{2}}} \Big)$$

$$= o(1)$$

for $i, j \in [M]\backslash\{1\}, n, \overline{n} \in [N], n \neq \overline{n}$. The first inequality is triangle inequality, the second inequality is by (205), the last equality is by $d_h = \widetilde{\Omega}\Big( \max\{\mathrm{SNR}^4, \mathrm{SNR}^{-4}\}N^2\epsilon^{-2}\Big)$ and $d =$

$\widetilde{\Omega}\Big(\epsilon^{-2}N^2 d_h\Big).$

$$|\langle \boldsymbol{k}_+^{(t+1)}, \boldsymbol{k}_-^{(t+1)}\rangle| \leq |\langle \boldsymbol{k}_+^{(T_2)}, \boldsymbol{k}_-^{(T_2)}\rangle| + \sum_{s=T_2}^{t}\left|\langle \boldsymbol{k}_+^{(s+1)}, \boldsymbol{k}_-^{(s+1)}\rangle - \langle \boldsymbol{k}_+^{(s)}, \boldsymbol{k}_-^{(s)}\rangle\right|$$

$$\leq |\langle \boldsymbol{k}_+^{(T_2)}, \boldsymbol{k}_-^{(T_2)}\rangle|$$

$$+ \sum_{s=T_2}^{t}\left|\beta_{+,+}^{(s)}\langle \boldsymbol{q}_+^{(s)}, \boldsymbol{k}_-^{(s)}\rangle + \sum_{n\in S_+}\sum_{i=2}^{M}\beta_{n,+,i}^{(s)}\langle \boldsymbol{q}_{n,i}^{(s)}, \boldsymbol{k}_-^{(s)}\rangle\right.$$

$$+ \beta_{-,-}^{(s)}\langle \boldsymbol{q}_-^{(s)}, \boldsymbol{k}_+^{(s)}\rangle + \sum_{n\in S_-}\sum_{i=2}^{M}\beta_{n,-,i}^{(s)}\langle \boldsymbol{q}_{n,i}^{(s)}, \boldsymbol{k}_+^{(s)}\rangle$$

$$+ \Big(\beta_{+,+}^{(s)}\boldsymbol{q}_+^{(s)} + \sum_{n\in S_+}\sum_{i=2}^{M}\beta_{n,+,i}^{(s)}\boldsymbol{q}_{n,i}^{(s)}\Big)$$

$$\left.\cdot\Big(\beta_{-,-}^{(s)}\boldsymbol{q}_-^{(s)\top} + \sum_{n\in S_-}\sum_{i=2}^{M}\beta_{n,-,i}^{(s)}\boldsymbol{q}_{n,i}^{(s)\top}\Big)\right|$$

$$\leq |\langle \boldsymbol{k}_+^{(T_2)}, \boldsymbol{k}_-^{(T_2)}\rangle|$$

$$+ \sum_{s=T_2}^{t}|\beta_{+,+}^{(s)}||\langle \boldsymbol{q}_+^{(s)}, \boldsymbol{k}_-^{(s)}\rangle| + \sum_{n\in S_+}\sum_{i=2}^{M}\sum_{s=T_2}^{t}|\beta_{n,+,i}^{(s)}||\langle \boldsymbol{q}_{n,i}^{(s)}, \boldsymbol{k}_-^{(s)}\rangle|$$

$$+ \sum_{s=T_2}^{t}|\beta_{-,-}^{(s)}||\langle \boldsymbol{q}_-^{(s)}, \boldsymbol{k}_+^{(s)}\rangle| + \sum_{n\in S_-}\sum_{i=2}^{M}\sum_{s=T_2}^{t}|\beta_{n,-,i}^{(s)}||\langle \boldsymbol{q}_{n,i}^{(s)}, \boldsymbol{k}_+^{(s)}\rangle|$$

$$+ \{lower\ order\ term\}$$

$$= |\langle \boldsymbol{k}_+^{(T_2)}, \boldsymbol{k}_-^{(T_2)}\rangle| + O\Big(\frac{N\big(\log(6N^2M^2/\delta)\big)^3 \log\big(O(\frac{1}{\epsilon})\big)}{\epsilon d_h^{\frac{1}{2}}}\Big)\cdot \log(\epsilon^{-1}d_h^{\frac{1}{2}})$$

$$+ N\cdot M\cdot O\Big(\frac{\mathrm{SNR}^2\big(\log(6N^2M^2/\delta)\big)^3 \log\big(O(\frac{1}{\epsilon})\big)}{\epsilon d_h^{\frac{1}{2}}}\Big)\cdot \log(\epsilon^{-1}d_h^{\frac{1}{2}})$$

$$= |\langle \boldsymbol{k}_+^{(T_2)}, \boldsymbol{k}_-^{(T_2)}\rangle| + O\Big(\frac{N\big(\log(6N^2M^2/\delta)\big)^3 \log\big(O(\frac{1}{\epsilon})\big)\log(\epsilon^{-1}d_h^{\frac{1}{2}})}{\epsilon d_h^{\frac{1}{2}}}\Big)$$

$$+ O\Big(\frac{N\cdot \mathrm{SNR}^2\big(\log(6N^2M^2/\delta)\big)^3 \log\big(O(\frac{1}{\epsilon})\big)\log(\epsilon^{-1}d_h^{\frac{1}{2}})}{\epsilon d_h^{\frac{1}{2}}}\Big)$$

$$= o(1),$$

where the first inequality is triangle inequality, the second inequality is by (214), the last equality is by $d_h = \widetilde{\Omega}\Big(\max\{\mathrm{SNR}^4, \mathrm{SNR}^{-4}\}N^2\epsilon^{-2}\Big).$

$$|\langle \boldsymbol{k}_+^{(t+1)}, \boldsymbol{k}_{n,i}^{(t+1)}\rangle| \leq |\langle \boldsymbol{k}_+^{(T_2)}, \boldsymbol{k}_{n,i}^{(T_2)}\rangle| + \sum_{s=T_2}^{t}\left|\langle \boldsymbol{k}_+^{(s+1)}, \boldsymbol{k}_{n,i}^{(s+1)}\rangle\right|$$

$$\leq |\langle \boldsymbol{k}_+^{(T_2)}, \boldsymbol{k}_{n,i}^{(T_2)}\rangle|$$

$$+ \sum_{s=T_2}^{t}\left|\beta_{+,+}^{(s)}\langle \boldsymbol{q}_+^{(s)}, \boldsymbol{k}_{n,i}^{(s)}\rangle + \sum_{n'\in S_+}\sum_{l=2}^{M}\beta_{n',+,l}^{(s)}\langle \boldsymbol{q}_{n',l}^{(s)}, \boldsymbol{k}_{n,i}^{(s)}\rangle\right.$$

$$+ \beta_{n,i,+}^{(s)}\langle \boldsymbol{q}_+^{(s)}, \boldsymbol{k}_+^{(s)}\rangle + \beta_{n,i,-}^{(s)}\langle \boldsymbol{q}_-^{(s)}, \boldsymbol{k}_+^{(s)}\rangle + \sum_{n'=1}^{N}\sum_{l=2}^{M}\beta_{n,i,n',l}^{(s)}\langle \boldsymbol{q}_{n',l}^{(s)}, \boldsymbol{k}_+^{(s)}\rangle$$

$$+ \left( \beta_{+,+}^{(s)} \boldsymbol{q}_+^{(s)} + \sum_{n' \in S_+} \sum_{l=2}^{M} \beta_{n',+,l}^{(s)} \boldsymbol{q}_{n',l}^{(s)} \right)$$

$$\cdot \left( \beta_{n,i,+}^{(s)} \boldsymbol{q}_+^{(s)\top} + \beta_{n,i,-}^{(s)} \boldsymbol{q}_-^{(s)\top} + \sum_{n'=1}^{N} \sum_{l=2}^{M} \beta_{n,i,n',l}^{(s)} \boldsymbol{q}_{n',l}^{(s)\top} \right) \Big|$$

$$\leq |\langle \boldsymbol{k}_+^{(T_2)}, \boldsymbol{k}_{n,i}^{(T_2)} \rangle|$$

$$+ \sum_{s=T_2}^{t} |\beta_{+,+}^{(s)}| |\langle \boldsymbol{q}_+^{(s)}, \boldsymbol{k}_{n,i}^{(s)} \rangle| + \sum_{l=2}^{M} \sum_{s=T_2}^{t} |\beta_{n,+,l}^{(s)}| |\langle \boldsymbol{q}_{n,l}^{(s)}, \boldsymbol{k}_{n,i}^{(s)} \rangle|$$

$$+ \sum_{n' \in S_+ \wedge n' \neq n} \sum_{l=2}^{M} \sum_{s=T_2}^{t} |\beta_{n',+,l}^{(s)}| |\langle \boldsymbol{q}_{n',l}^{(s)}, \boldsymbol{k}_{n,i}^{(s)} \rangle| + \sum_{s=T_2}^{t} |\beta_{n,i,+}^{(s)}| |\langle \boldsymbol{q}_+^{(s)}, \boldsymbol{k}_+^{(s)} \rangle|$$

$$+ \sum_{s=T_2}^{t} |\beta_{n,i,-}^{(s)}| |\langle \boldsymbol{q}_-^{(s)}, \boldsymbol{k}_+^{(s)} \rangle| + \sum_{l=2}^{M} \sum_{s=T_2}^{t} |\beta_{n,i,n,l}^{(s)}| |\langle \boldsymbol{q}_{n,l}^{(s)}, \boldsymbol{k}_+^{(s)} \rangle|$$

$$+ \sum_{n' \neq n}^{N} \sum_{l=2}^{M} \sum_{s=T_2}^{t} |\beta_{n,i,n',l}^{(s)}| |\langle \boldsymbol{q}_{n',l}^{(s)}, \boldsymbol{k}_+^{(s)} \rangle|$$

$$+ \{lower\ order\ term\}$$

$$= |\langle \boldsymbol{k}_+^{(T_2)}, \boldsymbol{k}_{n,i}^{(T_2)} \rangle|$$

$$+ O\left( \frac{N \left( \log(6N^2 M^2/\delta) \right)^3 \log \left( O(\frac{1}{\epsilon}) \right)}{\epsilon d_h^{\frac{1}{2}}} \right) \cdot \log(\epsilon^{-1} d_h^{\frac{1}{2}})$$

$$+ M \cdot O\left( \frac{\mathrm{SNR}^2 \left( \log(6N^2 M^2/\delta) \right)^3 \log \left( O(\frac{1}{\epsilon}) \right)}{\epsilon d_h^{\frac{1}{2}}} \right) \cdot \log(\epsilon^{-1} d_h^{\frac{1}{2}})$$

$$+ N \cdot M \cdot O\left( \frac{\mathrm{SNR}^2 \left( \log(6N^2 M^2/\delta) \right)^3 \log \left( O(\frac{1}{\epsilon}) \right)}{\epsilon d_h^{\frac{1}{2}}} \right) \cdot o(1)$$

$$+ O\left( \frac{N \left( \log(6N^2 M^2/\delta) \right)^3 \log \left( O(\frac{1}{\epsilon}) \right)}{\epsilon d_h^{\frac{1}{2}}} \right) \cdot \log(\epsilon^{-1} d_h^{\frac{1}{2}})$$

$$+ M \cdot O\left( \frac{\left( \log(6N^2 M^2/\delta) \right)^3 \log \left( O(\frac{1}{\epsilon}) \right)}{\epsilon d_h^{\frac{1}{2}}} \right) \cdot \log(\epsilon^{-1} d_h^{\frac{1}{2}})$$

$$+ N \cdot M \cdot O\left( \frac{\left( \log(6N^2 M^2/\delta) \right)^4 \log \left( O(\frac{1}{\epsilon}) \right)}{\epsilon d_h^{\frac{1}{2}} d_h^{\frac{1}{2}}} \right) \cdot \log(\epsilon^{-1} d_h^{\frac{1}{2}})$$

$$= |\langle \boldsymbol{k}_+^{(T_2)}, \boldsymbol{k}_{n,i}^{(T_2)} \rangle| + O\left( \frac{N \left( \log(6N^2 M^2/\delta) \right)^3 \log \left( O(\frac{1}{\epsilon}) \right) \log(\epsilon^{-1} d_h^{\frac{1}{2}})}{\epsilon d_h^{\frac{1}{2}}} \right)$$

$$+ o\left( \frac{N \cdot \mathrm{SNR}^2 \left( \log(6N^2 M^2/\delta) \right)^3 \log \left( O(\frac{1}{\epsilon}) \right)}{\epsilon d_h^{\frac{1}{2}}} \right)$$

$$= o(1)$$

where the first inequality is triangle inequality, the second inequality is by (209), the last equality is by $d_h = \widetilde{\Omega}\left( \max\{\mathrm{SNR}^4, \mathrm{SNR}^{-4}\} N^2 \epsilon^{-2} \right)$. Similarly, we have $|\langle \boldsymbol{k}_-^{(t+1)}, \boldsymbol{k}_{n,i}^{(t+1)} \rangle| = o(1)$.

$$|\langle \boldsymbol{k}_{n,i}^{(t+1)}, \boldsymbol{k}_{n,j}^{(t+1)} \rangle| \leq |\langle \boldsymbol{k}_{n,i}^{(T_2)}, \boldsymbol{k}_{n,j}^{(T_2)} \rangle| + \sum_{s=T_2}^{t} \left| \langle \boldsymbol{k}_{n,i}^{(s+1)}, \boldsymbol{k}_{n,j}^{(s+1)} \rangle - \langle \boldsymbol{k}_{n,i}^{(s)}, \boldsymbol{k}_{n,j}^{(s)} \rangle \right|$$

$$\leq |\langle \boldsymbol{k}_{n,i}^{(T_2)}, \boldsymbol{k}_{n,j}^{(T_2)} \rangle|$$

$$+ \sum_{s=T_2}^{t} \left| \beta_{n,i,+}^{(s)} \langle \boldsymbol{q}_+^{(s)}, \boldsymbol{k}_{n,j}^{(s)} \rangle + \beta_{n,i,-}^{(s)} \langle \boldsymbol{q}_-^{(s)}, \boldsymbol{k}_{n,j}^{(s)} \rangle + \sum_{n'=1}^{N} \sum_{l=2}^{M} \beta_{n,i,n',l}^{(s)} \langle \boldsymbol{q}_{n',l}^{(s)}, \boldsymbol{k}_{n,j}^{(s)} \rangle \right.$$

$$+ \beta_{n,j,+}^{(s)} \langle \boldsymbol{q}_+^{(s)}, \boldsymbol{k}_{n,i}^{(s)} \rangle + \beta_{n,j,-}^{(s)} \langle \boldsymbol{q}_-^{(s)}, \boldsymbol{k}_{n,i}^{(s)} \rangle + \sum_{n'=1}^{N} \sum_{l=2}^{M} \beta_{n,j,n',l}^{(s)} \langle \boldsymbol{q}_{n',l}^{(s)}, \boldsymbol{k}_{n,i}^{(s)} \rangle$$

$$+ \left( \beta_{n,i,+}^{(s)} \boldsymbol{q}_+^{(s)} + \beta_{n,i,-}^{(s)} \boldsymbol{q}_-^{(s)} + \sum_{n'=1}^{N} \sum_{l=2}^{M} \beta_{n,i,n',l}^{(s)} \boldsymbol{q}_{n',l}^{(s)} \right)$$

$$\left. \cdot \left( \beta_{n,j,+}^{(s)} \boldsymbol{q}_+^{(s)\top} + \beta_{n,j,-}^{(s)} \boldsymbol{q}_-^{(s)\top} + \sum_{n'=1}^{N} \sum_{l=2}^{M} \beta_{n,j,n',l}^{(s)} \boldsymbol{q}_{n',l}^{(s)\top} \right) \right|$$

$$\leq |\langle \boldsymbol{k}_{n,i}^{(T_2)}, \boldsymbol{k}_{n,j}^{(T_2)} \rangle|$$

$$+ \sum_{s=T_2}^{t} |\beta_{n,i,+}^{(s)}| |\langle \boldsymbol{q}_+^{(s)}, \boldsymbol{k}_{n,j}^{(s)} \rangle| + \sum_{s=T_2}^{t} |\beta_{n,i,-}^{(s)}| |\langle \boldsymbol{q}_-^{(s)}, \boldsymbol{k}_{n,j}^{(s)} \rangle| + \sum_{l=2}^{M} \sum_{s=T_2}^{t} |\beta_{n,i,n,l}^{(s)}| |\langle \boldsymbol{q}_{n,l}^{(s)}, \boldsymbol{k}_{n,j}^{(s)} \rangle|$$

$$+ \sum_{n' \neq n}^{N} \sum_{l=2}^{M} \sum_{s=T_2}^{t} |\beta_{n,i,n',l}^{(s)}| |\langle \boldsymbol{q}_{n',l}^{(s)}, \boldsymbol{k}_{n,j}^{(s)} \rangle| + \sum_{s=T_2}^{t} |\beta_{n,j,+}^{(s)}| |\langle \boldsymbol{q}_+^{(s)}, \boldsymbol{k}_{n,i}^{(s)} \rangle| + \sum_{s=T_2}^{t} |\beta_{n,j,-}^{(s)}| |\langle \boldsymbol{q}_-^{(s)}, \boldsymbol{k}_{n,i}^{(s)} \rangle|$$

$$+ \sum_{l=2}^{M} \sum_{s=T_2}^{t} |\beta_{n,j,n,l}^{(s)}| |\langle \boldsymbol{q}_{n,l}^{(s)}, \boldsymbol{k}_{n,i}^{(s)} \rangle| + \sum_{n' \neq n}^{N} \sum_{l=2}^{M} \sum_{s=T_2}^{t} |\beta_{n,j,n',l}^{(s)}| |\langle \boldsymbol{q}_{n',l}^{(s)}, \boldsymbol{k}_{n,i}^{(s)} \rangle|$$

$$+ \{lower\ order\ term\}$$

$$= |\langle \boldsymbol{k}_{n,i}^{(T_2)}, \boldsymbol{k}_{n,j}^{(T_2)} \rangle|$$

$$+ O\left( \frac{N \left( \log(6N^2 M^2/\delta) \right)^3 \log \left( O(\frac{1}{\epsilon}) \right)}{\epsilon d_h^{\frac{1}{2}}} \right) \cdot \log(\epsilon^{-1} d_h^{\frac{1}{2}})$$

$$+ M \cdot O\left( \frac{\left( \log(6N^2 M^2/\delta) \right)^3 \log \left( O(\frac{1}{\epsilon}) \right)}{\epsilon d_h^{\frac{1}{2}}} \right) \cdot \log(\epsilon^{-1} d_h^{\frac{1}{2}})$$

$$+ N \cdot M \cdot O\left( \frac{\left( \log(6N^2 M^2/\delta) \right)^4 \log \left( O(\frac{1}{\epsilon}) \right)}{\epsilon d^{\frac{1}{2}} d_h^{\frac{1}{2}}} \right) \cdot o(1)$$

$$= |\langle \boldsymbol{k}_{n,i}^{(T_2)}, \boldsymbol{k}_{n,j}^{(T_2)} \rangle| + O\left( \frac{N \left( \log(6N^2 M^2/\delta) \right)^3 \log \left( O(\frac{1}{\epsilon}) \right) \log(\epsilon^{-1} d_h^{\frac{1}{2}})}{\epsilon d_h^{\frac{1}{2}}} \right)$$

$$+ o\left( \frac{N \left( \log(6N^2 M^2/\delta) \right)^4 \log \left( O(\frac{1}{\epsilon}) \right)}{\epsilon d^{\frac{1}{2}} d_h^{\frac{1}{2}}} \right)$$

$$= o(1)$$

for $i, j \in [M] \setminus \{1\}, i \neq j, n \in [N]$. The first inequality is triangle inequality, the second inequality is by (211), the last equality is by $d_h = \widetilde{\Omega}\left( \max\{\text{SNR}^4, \text{SNR}^{-4}\} N^2 \epsilon^{-2} \right)$ and $d = \widetilde{\Omega}\left( \epsilon^{-2} N^2 d_h \right)$.

$$|\langle \boldsymbol{k}_{n,i}^{(t+1)}, \boldsymbol{k}_{\overline{n},j}^{(t+1)} \rangle| \leq |\langle \boldsymbol{k}_{n,i}^{(T_2)}, \boldsymbol{k}_{\overline{n},j}^{(T_2)} \rangle| + \sum_{s=T_2}^{t} \left| \langle \boldsymbol{k}_{n,i}^{(s+1)}, \boldsymbol{k}_{\overline{n},j}^{(s+1)} \rangle - \langle \boldsymbol{k}_{n,i}^{(s)}, \boldsymbol{k}_{\overline{n},j}^{(s)} \rangle \right|$$

$$\leq |\langle \boldsymbol{k}_{n,i}^{(T_2)}, \boldsymbol{k}_{\overline{n},j}^{(T_2)} \rangle|$$

$$+ \sum_{s=T_2}^{t} \left| \beta_{n,i,+}^{(s)} \langle \boldsymbol{q}_+^{(s)}, \boldsymbol{k}_{\overline{n},j}^{(s)} \rangle + \beta_{n,i,-}^{(s)} \langle \boldsymbol{q}_-^{(s)}, \boldsymbol{k}_{\overline{n},j}^{(s)} \rangle + \sum_{n'=1}^{N} \sum_{l=2}^{M} \beta_{n,i,n',l}^{(s)} \langle \boldsymbol{q}_{n',l}^{(s)}, \boldsymbol{k}_{\overline{n},j}^{(s)} \rangle \right.$$

$$+ \beta_{\overline{n},j,+}^{(s)} \langle \boldsymbol{q}_+^{(s)}, \boldsymbol{k}_{n,i}^{(s)} \rangle + \beta_{\overline{n},j,-}^{(s)} \langle \boldsymbol{q}_-^{(s)}, \boldsymbol{k}_{n,i}^{(s)} \rangle + \sum_{n'=1}^{N} \sum_{l=2}^{M} \beta_{\overline{n},j,n',l}^{(s)} \langle \boldsymbol{q}_{n',l}^{(s)}, \boldsymbol{k}_{n,i}^{(s)} \rangle$$

$$+ \left( \beta_{n,i,+}^{(s)} \boldsymbol{q}_+^{(s)} + \beta_{n,i,-}^{(s)} \boldsymbol{q}_-^{(s)} + \sum_{n'=1}^{N} \sum_{l=2}^{M} \beta_{n,i,n',l}^{(s)} \boldsymbol{q}_{n',l}^{(s)} \right)$$

$$\cdot \left( \beta_{\overline{n},j,+}^{(s)} \boldsymbol{q}_+^{(s)\top} + \beta_{\overline{n},j,-}^{(s)} \boldsymbol{q}_-^{(s)\top} + \sum_{n'=1}^{N} \sum_{l=2}^{M} \beta_{\overline{n},j,n',l}^{(s)} \boldsymbol{q}_{n',l}^{(s)\top} \right) \Big|$$

$$\leq |\langle \boldsymbol{k}_{n,i}^{(T_2)}, \boldsymbol{k}_{\overline{n},j}^{(T_2)} \rangle|$$

$$+ \sum_{s=T_2}^{t} |\beta_{n,i,+}^{(s)}| |\langle \boldsymbol{q}_+^{(s)}, \boldsymbol{k}_{\overline{n},j}^{(s)} \rangle| + \sum_{s=T_2}^{t} |\beta_{n,i,-}^{(s)}| |\langle \boldsymbol{q}_-^{(s)}, \boldsymbol{k}_{\overline{n},j}^{(s)} \rangle| + \sum_{l=2}^{M} \sum_{s=T_2}^{t} |\beta_{n,i,\overline{n},l}^{(s)}| |\langle \boldsymbol{q}_{\overline{n},l}^{(s)}, \boldsymbol{k}_{\overline{n},j}^{(s)} \rangle|$$

$$+ \sum_{l=2}^{M} \sum_{s=T_2}^{t} |\beta_{n,i,n,l}^{(s)}| |\langle \boldsymbol{q}_{n,l}^{(s)}, \boldsymbol{k}_{\overline{n},j}^{(s)} \rangle| + \sum_{n' \neq n \wedge n' \overline{n}} \sum_{l=2}^{M} \sum_{s=T_2}^{t} |\beta_{n,i,n',l}^{(s)}| |\langle \boldsymbol{q}_{n',l}^{(s)}, \boldsymbol{k}_{\overline{n},j}^{(s)} \rangle|$$

$$+ \sum_{s=T_2}^{t} |\beta_{\overline{n},j,+}^{(s)}| |\langle \boldsymbol{q}_+^{(s)}, \boldsymbol{k}_{n,i}^{(s)} \rangle| + \sum_{s=T_2}^{t} |\beta_{\overline{n},j,-}^{(s)}| |\langle \boldsymbol{q}_-^{(s)}, \boldsymbol{k}_{n,i}^{(s)} \rangle| + \sum_{l=2}^{M} \sum_{s=T_2}^{t} |\beta_{\overline{n},j,n,l}^{(s)}| |\langle \boldsymbol{q}_{n,l}^{(s)}, \boldsymbol{k}_{n,i}^{(s)} \rangle|$$

$$+ \sum_{l=2}^{M} \sum_{s=T_2}^{t} |\beta_{\overline{n},j,\overline{n},l}^{(s)}| |\langle \boldsymbol{q}_{\overline{n},l}^{(s)}, \boldsymbol{k}_{n,i}^{(s)} \rangle| + \sum_{n' \neq n \wedge n' \overline{n}} \sum_{l=2}^{M} \sum_{s=T_2}^{t} |\beta_{\overline{n},j,n',l}^{(s)}| |\langle \boldsymbol{q}_{n',l}^{(s)}, \boldsymbol{k}_{n,i}^{(s)} \rangle|$$

$$+ \{lower\ order\ term\}$$

$$= |\langle \boldsymbol{k}_{n,i}^{(T_2)}, \boldsymbol{k}_{\overline{n},j}^{(T_2)} \rangle|$$

$$+ O\left( \frac{N \left( \log(6N^2 M^2/\delta) \right)^3 \log \left( O(\frac{1}{\epsilon}) \right)}{\epsilon d_h^{\frac{1}{2}}} \right) \cdot \log(\epsilon^{-1} d_h^{\frac{1}{2}}) + M \cdot O\left( \frac{\left( \log(6N^2 M^2/\delta) \right)^4 \log \left( O(\frac{1}{\epsilon}) \right)}{\epsilon d^{\frac{1}{2}} d_h^{\frac{1}{2}}} \right) \cdot \log(\epsilon^{-1} d_h^{\frac{1}{2}})$$

$$+ M \cdot O\left( \frac{\left( \log(6N^2 M^2/\delta) \right)^3 \log \left( O(\frac{1}{\epsilon}) \right)}{\epsilon d_h^{\frac{1}{2}}} \right) \cdot o(1) + N \cdot M \cdot O\left( \frac{\left( \log(6N^2 M^2/\delta) \right)^4 \log \left( O(\frac{1}{\epsilon}) \right)}{\epsilon d^{\frac{1}{2}} d_h^{\frac{1}{2}}} \right) \cdot o(1)$$

$$= |\langle \boldsymbol{k}_{n,i}^{(T_2)}, \boldsymbol{k}_{\overline{n},j}^{(T_2)} \rangle|$$

$$+ O\left( \frac{N \left( \log(6N^2 M^2/\delta) \right)^3 \log \left( O(\frac{1}{\epsilon}) \right) \log(\epsilon^{-1} d_h^{\frac{1}{2}})}{\epsilon d_h^{\frac{1}{2}}} \right) + o\left( \frac{N \left( \log(6N^2 M^2/\delta) \right)^4 \log \left( O(\frac{1}{\epsilon}) \right)}{\epsilon d^{\frac{1}{2}} d_h^{\frac{1}{2}}} \right)$$

$$= o(1)$$

for $i, j \in [M] \backslash \{1\}, n, \overline{n} \in [N], n \neq \overline{n}$. The first inequality is triangle inequality, the second inequality is by (212), the last equality is by $d_h = \widetilde{\Omega}\left( \max\{\text{SNR}^4, \text{SNR}^{-4}\} N^2 \epsilon^{-2} \right)$ and $d = \widetilde{\Omega}\left( \epsilon^{-2} N^2 d_h \right)$.

### F.12 Explanations of Lower Order Terms

In this section, we provide some explanations of lower order terms to demonstrate the rigor of our proof.

To bound the so-call $\{lower\ order\ term\}$, we condition that dimensions $d$, $d_h$ are sufficiently large and learning rate $\eta$ is sufficiently small. Next, we show how we utilize these three parameters.

**Sufficiently mall learning rate $\eta$ :**   Recall the dynamics of QK

$$\langle \boldsymbol{q}^{(t+1)}, \boldsymbol{k}^{(t+1)} \rangle - \langle \boldsymbol{q}^{(t)}, \boldsymbol{k}^{(t)} \rangle = \langle \Delta \boldsymbol{q}^{(t)}, \boldsymbol{k}^{(t)} \rangle + \langle \boldsymbol{q}^{(t)}, \Delta \boldsymbol{k}^{(t)} \rangle + \langle \Delta \boldsymbol{q}^{(t)}, \Delta \boldsymbol{k}^{(t)} \rangle$$

Note that terms $\langle \Delta \boldsymbol{q}^{(t)}, \boldsymbol{k}^{(t)} \rangle$, $\langle \boldsymbol{q}^{(t)}, \Delta \boldsymbol{k}^{(t)} \rangle$ contain factor $\eta$, and term $\langle \Delta \boldsymbol{q}^{(t)}, \Delta \boldsymbol{k}^{(t)} \rangle$ contains factor $\eta^2$. Therefore, as long as $\eta$ is sufficiently small, $\langle \Delta \boldsymbol{q}^{(t)}, \Delta \boldsymbol{k}^{(t)} \rangle$ is sufficiently small than $\langle \Delta \boldsymbol{q}^{(t)}, \boldsymbol{k}^{(t)} \rangle$ and $\langle \boldsymbol{q}^{(t)}, \Delta \boldsymbol{k}^{(t)} \rangle$. Now we take the dynamic of $\langle \boldsymbol{q}_+^{(t)}, \boldsymbol{k}_+^{(t)} \rangle$ as an example.

$$\langle \boldsymbol{q}_+^{(t+1)}, \boldsymbol{k}_+^{(t+1)} \rangle - \langle \boldsymbol{q}_+^{(t)}, \boldsymbol{k}_+^{(t)} \rangle$$

$$= \alpha_{+,+}^{(t)} \|\boldsymbol{k}_+^{(t)}\|_2^2 + \sum_{n \in S_+} \sum_{i=2}^{M} \alpha_{n,+,i}^{(t)} \langle \boldsymbol{k}_+^{(t)}, \boldsymbol{k}_{n,i}^{(t)} \rangle$$

$$+ \beta_{+,+}^{(t)} \|\boldsymbol{q}_+^{(t)}\|_2^2 + \sum_{n \in S_+} \sum_{i=2}^{M} \beta_{n,+,i}^{(t)} \langle \boldsymbol{q}_+^{(t)}, \boldsymbol{q}_{n,i}^{(t)} \rangle$$

$$+ \left( \alpha_{+,+}^{(t)} \boldsymbol{k}_+^{(t)} + \sum_{n \in S_+} \sum_{i=2}^{M} \alpha_{n,+,i}^{(t)} \boldsymbol{k}_{n,i}^{(t)} \right)$$

$$\cdot \left( \beta_{+,+}^{(t)} \boldsymbol{q}_+^{(t)\top} + \sum_{n \in S_+} \sum_{i=2}^{M} \beta_{n,+,i}^{(t)} \boldsymbol{q}_{n,i}^{(t)\top} \right),$$

Under benign overfitting regime, we have $\|\boldsymbol{k}_+^{(t)}\|_2^2 = \Theta(\|\boldsymbol{\mu}\|_2^2 \sigma_h^2 d_h)$, $\langle \boldsymbol{q}_+^{(t)}, \boldsymbol{k}_+^{(t)} \rangle \leq \log(\epsilon^{-1} d_h^{\frac{1}{2}})$ and $\beta_{+,+}^{(t)} = O(\eta \|\boldsymbol{\mu}\|_2^2)$. Therefore, as long as $\eta = o(\sigma_h^2 d_h (\log(\epsilon^{-1} d_h^{\frac{1}{2}}))^{-1})$, we have

$$\left( \alpha_{+,+}^{(t)} \boldsymbol{k}_+^{(t)} \right) \left( \beta_{+,+}^{(t)} \boldsymbol{q}_+^{(t)\top} \right) = o(\alpha_{+,+}^{(t)} \|\boldsymbol{k}_+^{(t)}\|_2^2).$$

Similar method can be used for other items in $\left( \alpha_{+,+}^{(t)} \boldsymbol{k}_+^{(t)} + \sum_{n \in S_+} \sum_{i=2}^{M} \alpha_{n,+,i}^{(t)} \boldsymbol{k}_{n,i}^{(t)} \right) \cdot \left( \beta_{+,+}^{(t)} \boldsymbol{q}_+^{(t)\top} + \sum_{n \in S_+} \sum_{i=2}^{M} \beta_{n,+,i}^{(t)} \boldsymbol{q}_{n,i}^{(t)\top} \right)$. At last, we have

$$\langle \boldsymbol{q}_+^{(t+1)}, \boldsymbol{k}_+^{(t+1)} \rangle - \langle \boldsymbol{q}_+^{(t)}, \boldsymbol{k}_+^{(t)} \rangle$$

$$= \alpha_{+,+}^{(t)} \|\boldsymbol{k}_+^{(t)}\|_2^2 + \sum_{n \in S_+} \sum_{i=2}^{M} \alpha_{n,+,i}^{(t)} \langle \boldsymbol{k}_+^{(t)}, \boldsymbol{k}_{n,i}^{(t)} \rangle$$

$$+ \beta_{+,+}^{(t)} \|\boldsymbol{q}_+^{(t)}\|_2^2 + \sum_{n \in S_+} \sum_{i=2}^{M} \beta_{n,+,i}^{(t)} \langle \boldsymbol{q}_+^{(t)}, \boldsymbol{q}_{n,i}^{(t)} \rangle \tag{297}$$

$$+ \{lower\ order\ term\}.$$

**Sufficiently large dimension $d_h$ :** Take (297) as an example, Noting that $\|\boldsymbol{k}_+^{(t)}\|_2^2 = O(\eta \|\boldsymbol{\mu}\|_2^2)$ and $\langle \boldsymbol{k}_+^{(t)}, \boldsymbol{k}_{n,i}^{(t)} \rangle = o(1)$. Therefore, as long as $d_h$ is sufficiently large, $\|\boldsymbol{k}_+^{(t)}\|_2^2$ is much larger than $\langle \boldsymbol{k}_+^{(t)}, \boldsymbol{k}_{n,i}^{(t)} \rangle$. Besides, by the property that the sum of each row and column of matrix $(diag(\boldsymbol{\varphi}_{n,1}^{(t)}) - \boldsymbol{\varphi}_{n,1}^{(t)\top} \boldsymbol{\varphi}_{n,1}^{(t)})$ is 0, we have $\alpha_{+,+}^{(t)} + \sum_{n \in S_+} \sum_{i=2}^{M} \alpha_{n,+,i}^{(t)} = 0$. We also prove $\alpha_{+,+}^{(t)} \geq 0$ and $\alpha_{n,+,i}^{(t)} \leq 0$ under benign overfitting regime, thus the magnitude of $\alpha_{n,+,i}^{(t)}$ is smaller than $\alpha_{+,+}^{(t)}$. All in all, it can be proved that $\sum_{n \in S_+} \sum_{i=2}^{M} \alpha_{n,+,i}^{(t)} \langle \boldsymbol{k}_+^{(t)}, \boldsymbol{k}_{n,i}^{(t)} \rangle = o(\alpha_{+,+}^{(t)} \|\boldsymbol{k}_+^{(t)}\|_2^2)$. This method can be further applied to $\beta_{+,+}^{(t)} \|\boldsymbol{q}_+^{(t)}\|_2^2 + \sum_{n \in S_+} \sum_{i=2}^{M} \beta_{n,+,i}^{(t)} \langle \boldsymbol{q}_+^{(t)}, \boldsymbol{q}_{n,i}^{(t)} \rangle$. At last, we can simplify (297) as follows:

$$\langle \boldsymbol{q}_+^{(t+1)}, \boldsymbol{k}_+^{(t+1)} \rangle - \langle \boldsymbol{q}_+^{(t)}, \boldsymbol{k}_+^{(t)} \rangle$$
$$= \alpha_{+,+}^{(t)} \|\boldsymbol{k}_+^{(t)}\|_2^2 + \beta_{+,+}^{(t)} \|\boldsymbol{q}_+^{(t)}\|_2^2 + \{lower\ order\ term\}. \tag{298}$$

**Sufficiently large dimension** $d$ **:** Take $\alpha_{n',i',+}^{(t)}$ as an example:

$$
\begin{aligned}
\alpha_{n',i',+}^{(t)} = \frac{\eta}{NM} \sum_{n \in S_+} -\ell_n'^{(t)} \sum_{i=2}^{M} \langle \boldsymbol{\xi}_{n',i'}, \boldsymbol{\xi}_{n,i} \rangle \\
\cdot \Bigg( V_+^{(t)} \Bigg( \frac{\exp(\langle \boldsymbol{q}_{n,i}^{(t)}, \boldsymbol{k}_+^{(t)} \rangle)}{\exp(\langle \boldsymbol{q}_{n,i}^{(t)}, \boldsymbol{k}_+^{(t)} \rangle) + \sum_{j=2}^{M} \exp(\langle \boldsymbol{q}_{n,i}^{(t)}, \boldsymbol{k}_{n,j}^{(t)} \rangle)} \\
- \Big( \frac{\exp(\langle \boldsymbol{q}_{n,i}^{(t)}, \boldsymbol{k}_+^{(t)} \rangle)}{\exp(\langle \boldsymbol{q}_{n,i}^{(t)}, \boldsymbol{k}_+^{(t)} \rangle) + \sum_{j=2}^{M} \exp(\langle \boldsymbol{q}_{n,i}^{(t)}, \boldsymbol{k}_{n,j}^{(t)} \rangle)} \Big)^2 \Big) \\
- \sum_{k=2}^{M} \Big( V_{n,i}^{(t)} \cdot \frac{\exp(\langle \boldsymbol{q}_{n,i}^{(t)}, \boldsymbol{k}_+^{(t)} \rangle)}{\exp(\langle \boldsymbol{q}_{n,i}^{(t)}, \boldsymbol{k}_+^{(t)} \rangle) + \sum_{j=2}^{M} \exp(\langle \boldsymbol{q}_{n,i}^{(t)}, \boldsymbol{k}_{n,j}^{(t)} \rangle)} \\
\cdot \frac{\exp(\langle \boldsymbol{q}_{n,i}^{(t)}, \boldsymbol{k}_{n,k}^{(t)} \rangle)}{\exp(\langle \boldsymbol{q}_{n,i}^{(t)}, \boldsymbol{k}_+^{(t)} \rangle) + \sum_{j=2}^{M} \exp(\langle \boldsymbol{q}_{n,i}^{(t)}, \boldsymbol{k}_{n,j}^{(t)} \rangle)} \Big) \Bigg),
\end{aligned}
\tag{299}
$$

Note that $\langle \boldsymbol{\xi}_{n',i'}, \boldsymbol{\xi}_{n,i} \rangle$ can be divided into two types: $\|\boldsymbol{\xi}_{n',i'}\|_2^2$ and $\langle \boldsymbol{\xi}_{n',i'}, \boldsymbol{\xi}_{n,i} \rangle$ for $i \neq i'$ or $n \neq n'$. By Lemma C.4, we have

$$
\tilde{\sigma}_p^2 d / 2 \leq \|\boldsymbol{\xi}_{n,2}\|_2^2 \leq 3\tilde{\sigma}_p^2 d / 2,
$$
$$
\sigma_p^2 d / 2 \leq \|\boldsymbol{\xi}_{n,i}\|_2^2 \leq 3\sigma_p^2 d / 2,
$$
$$
|\langle \boldsymbol{\xi}_{n,i}, \boldsymbol{\xi}_{n',i'} \rangle| \leq 2\tilde{\sigma}_p^2 \cdot \sqrt{d \log(4N^2 M^2 / \delta)}
$$

for $i, i' \in [M] \backslash \{1\}, n, n' \in [N], i \neq i'$ or $n \neq n'$.

As long as $d$ is sufficiently large, $\|\boldsymbol{\xi}_{n',i'}\|_2^2$ is much larger than $\langle \boldsymbol{\xi}_{n',i'}, \boldsymbol{\xi}_{n,i} \rangle$ for $i \neq i'$ or $n \neq n'$. Therefore, (299) can be further simplified as follows:

$$
\begin{aligned}
\alpha_{n',i',+}^{(t)} = -\frac{\eta}{NM} \ell_{n'}'^{(t)} \|\boldsymbol{\xi}_{n',i'}\|_2^2 \\
\cdot \Bigg( V_+^{(t)} \Bigg( \frac{\exp(\langle \boldsymbol{q}_{n',i'}^{(t)}, \boldsymbol{k}_+^{(t)} \rangle)}{\exp(\langle \boldsymbol{q}_{n',i'}^{(t)}, \boldsymbol{k}_+^{(t)} \rangle) + \sum_{j=2}^{M} \exp(\langle \boldsymbol{q}_{n',i'}^{(t)}, \boldsymbol{k}_{n',j}^{(t)} \rangle)} \\
- \Big( \frac{\exp(\langle \boldsymbol{q}_{n',i'}^{(t)}, \boldsymbol{k}_+^{(t)} \rangle)}{\exp(\langle \boldsymbol{q}_{n',i'}^{(t)}, \boldsymbol{k}_+^{(t)} \rangle) + \sum_{j=2}^{M} \exp(\langle \boldsymbol{q}_{n',i'}^{(t)}, \boldsymbol{k}_{n',j}^{(t)} \rangle)} \Big)^2 \Big) \\
- \sum_{k=2}^{M} \Big( V_{n',i'}^{(t)} \cdot \frac{\exp(\langle \boldsymbol{q}_{n',i'}^{(t)}, \boldsymbol{k}_+^{(t)} \rangle)}{\exp(\langle \boldsymbol{q}_{n',i'}^{(t)}, \boldsymbol{k}_+^{(t)} \rangle) + \sum_{j=2}^{M} \exp(\langle \boldsymbol{q}_{n',i'}^{(t)}, \boldsymbol{k}_{n',j}^{(t)} \rangle)} \\
\cdot \frac{\exp(\langle \boldsymbol{q}_{n',i'}^{(t)}, \boldsymbol{k}_{n',k}^{(t)} \rangle)}{\exp(\langle \boldsymbol{q}_{n',i'}^{(t)}, \boldsymbol{k}_+^{(t)} \rangle) + \sum_{j=2}^{M} \exp(\langle \boldsymbol{q}_{n',i'}^{(t)}, \boldsymbol{k}_{n',j}^{(t)} \rangle)} \Big) \Bigg) \\
+ \{lower\ order\ term\}.
\end{aligned}
\tag{300}
$$

# G   Takeaways for Practitioners

Our theoretical results mainly focuse on the impact of different $N$ and SNR on the generalization performance. So we can provide guidance from the perspective of increasing $N$, SNR and $N \cdot \text{SNR}^2$. The following are some practical scenarios

**Data Augmentation:** Researchers sometimes employ the technique of data augmentation by introducing controlled noise into their datasets. From the perspective of our paper's results, this method reduce SNR but improve N because we generate "new" data point by adding noises. As reducing SNR may be harmful to generalization performance, we must make sure that we use enough data points to train the model( enough sample size $N$ ).

**Semi-Supervised Learning:** Semi-supervised learning is useful when you have a small amount of labeled data and a large amount of unlabeled data. Labeled data can be seen as data with high SNR, while unlabeled data with low SNR because for some unlabeled samples, we may mistake their labels, making them equivalent to noises. In this scenario, we need to ensure that we have sufficient unlabeled data (enough sample size $N$) and make full use of labeled data (high SNR data points).

Overall, we need to consider both the sample size $N$ and the signal-to-noise ratio SNR to train the model.

# H   Broader Impacts

This work focus on theoretically studying the training dynamics and generalization of Transformer in Vision. The techniques used in this paper may be generalized to study other abilities of Transformer or other network models. Besides, the theoretical results in this paper may inspire more attempts at training large foundational models with high-quality data. We do not foresee any form of negative social impact induced by our work.

