# OpenReview forum: "Unveil Benign Overfitting for Transformer in Vision: Training Dynamics, Convergence, and Generalization"
_NeurIPS.cc/2024/Conference — NeurIPS 2024 poster_

### Official Review · Reviewer_4uNF · 2024-07-08

**Soundness:** 4
**Presentation:** 4
**Contribution:** 3
**Rating:** 5
**Confidence:** 3

**Summary:**

The paper investigates the benign overfitting phenomenon in Vision Transformers. By examining the training dynamics and generalization of a two-layer Transformer model, it establishes a condition to differentiate between benign and harmful overfitting based on the signal-to-noise ratio in the data model. Theoretical results are supported by experimental simulations.

**Strengths:**

1. The paper provides a deep theoretical understanding of how Vision Transformers can achieve benign overfitting, filling a gap in the literature.
2. The derivation of the conditions for benign and harmful overfitting is mathematically rigorous and well-founded. Strong conclusions drawn based on appropriate assumptions.
3. The theoretical findings are supported by experimental simulations, which confirm the sharp condition separating benign and harmful overfitting.

**Weaknesses:**

1. The sparsity assumption is too strong, as it relies on the signal being contained within one patch and the noise being contained within another, which may not align with real-world data distributions.
2. Experiments are limited to settings that perfectly align with theoretical assumptions. The paper should also explore scenarios that deviate from these assumptions to understand the limitations of the theoretical results.

**Questions:**

The authors are suggested to explore scenarios that deviate from these assumptions to understand the limitations of the theoretical results. For example, more signal patches and more noise patches are contained in all patches. Tests on real-world tasks are better.

**Limitations:**

See Questions and Weaknesses.

---

> ### Author Rebuttal · Authors · 2024-08-05
>
> Thanks for your constructive feedback! We address your questions and concerns as follows.
>
> ---
>
> **Q1**. The sparsity assumption is too strong, as it relies on the signal being contained within one patch and the noise being contained within another, which may not align with real-world data distributions.
>
> **A1**. The data generation model in this paper follows the common assumptions made in previous studies of benign overfitting ( Cao et al. (2022) [1], Kou et al. (2023) [2]). This assumption is not so strong for recent studies on benign overfitting, especially when we investigate on more complex ViT models. As Reviewer UpG9 says, "To the best of my knowledge, this is the first work to successfully address the learning dynamics of even a simple transformer architecture without unrealistic assumptions".
>
> We also acknowledge that the data generation model may not align perfectly with real-world data distributions. In our perspective, the signals in the data model simulate the targets within an image, while the noises simulate the background in the image. We perform some experiments on MNIST dataset to verify our theoretical results (see next question), and we will try to make our theoretical results helpful for more practical applications.
>
> ---
>
> **Q2**. The authors are suggested to explore scenarios that deviate from these assumptions to understand the limitations of the theoretical results. For example, more signal patches and more noise patches are contained in all patches. Tests on real-world tasks are better.
>
> **A2**. We perform some experiments on MNIST dataset. In these experiments, we used different dataset, different noise, and different network model than in this paper. So the results regarding N and SNR may be different, which shows the limitations of our theoretical results.
>
> One of our important conclusion is valid, that is, the larger the sample size $N$ and signal-to-noise ratio SNR, the better the generalization performance. Please refer to the uploaded PDF file.
>
> ---
>
> Reference
>
> [1] Cao, Y., Chen,Z., Belkin,M., and Gu,Q. Benign overfitting in two-layer convolutional neural networks. In NeurIPS 2022
>
> [2] Kou, Y., Chen, Z., Chen, Y., and Gu, Q. Benign overfitting in two-layer ReLU convolutional neural networks. In ICML 2023

---

> ### Author Response · Authors · 2024-08-13
>
> It will be very encouraging that you can reconsider raising your score if we have addressed all the issues you raised. Otherwise, we are happy with the further discussion.

---

### Official Review · Reviewer_ysmG · 2024-07-08

**Soundness:** 3
**Presentation:** 3
**Contribution:** 2
**Rating:** 5
**Confidence:** 1

**Summary:**

This paper provides a sharp theoretical characterization of the transition between benign and harmful overfitting regimes for Vision Transformers trained on linearly separable data. The authors carefully analyze the optimization dynamics and provide generalization bounds that depend on the signal-to-noise ratio of the data. Extensive experiments validate the theory.

**Strengths:**

* Provides a precise characterization of benign vs harmful overfitting regimes for Vision Transformers
* Writing is clear and easy to follow
* Novel results on the harmful overfitting regime
* Extensive experiments align well with and validate the theory

**Weaknesses:**

* Only considers the linearly separable setting, which is already solvable by existing vision and language models, so the conclusions are not very surprising even if initialization and model details differ
* Lacks clear takeaways for practitioners - how can these theoretical insights be used to improve real-world vision models?
* Could the authors provide a more rigorous perspective on transformer bias (e.g. low rank structure) from an optimization perspective under their model?

**Questions:**

* If an MLP with ReLU activation was used instead of the simple linearly separable task, would the optimization dynamics converge faster or slower? Would the SNR requirements be stricter or more relaxed?
* What guidance can the authors provide to practitioners based on these theoretical results?

**Limitations:**

The authors acknowledge focusing only on the simplified linearly separable setting. More discussion on how the insights could extend to real-world nonlinear problems would be valuable.

---

> ### Author Rebuttal · Authors · 2024-08-05
>
> Thanks for your constructive feedback! We address your questions and concerns as follows.
>
> ---
>
> **Q1**. Only considers the linearly separable setting, which is already solvable by existing vision and language models, so the conclusions are not very surprising even if initialization and model details differ.
>
> **A1**. We follows the common assumptions made in previous studies of benign overfitting ( Cao et al. (2022)[1], Kou et al. (2023)[2] ) for the data generation model in this paper. Although this problem seems simple, rigorously proving these conclusions to show benign overfitting phenomenon is very challenging when considering the complexity of ViTs. In addition, rather than only presenting benign overfitting conclusion as most existing works, we also present harmful overfitting conclusion to show that the ViT cannot even learn a linearly separable dataset when the sample size $N$ and signal-to-noise ratio SNR is low.
>
> ---
>
> **Q2**. Lacks clear takeaways for practitioners. What guidance can the authors provide to practitioners based on these theoretical results?
>
> **A2**. Our theoretical results mainly focuse on the impact of different $N$ and SNR on the generalization performance. So we can provide guidance from the perspective of increasing $N$, SNR and $N \cdot \mathrm{SNR}^2$. The following are some practical scenarios.
>
> - Data Augmentation: Researchers sometimes employ the technique of data augmentation by introducing controlled noise into their datasets. From the perspective of our paper's results, this method reduce SNR but improve N because we generate "new" data point by adding noises. As reducing SNR may be harmful to generalization performance, we must make sure that we use enough data points to train the model( enough sample size $N$ ).
> - Semi-Supervised Learning: Semi-supervised learning is useful when you have a small amount of labeled data and a large amount of unlabeled data. Labeled data can be seen as data with high SNR, while unlabeled data with low SNR because for some unlabeled samples, we may mistake their labels, making them equivalent to noises. In this scenario, we need to ensure that we have sufficient unlabeled data (enough sample size $N$) and make full use of labeled data (high SNR data points).
>
> Overall, we need to consider both the sample size $N$ and the signal-to-noise ratio SNR to train the model.
>
> ---
>
> **Q3**. Could the authors provide a more rigorous perspective on transformer bias (e.g. low rank structure) from an optimization perspective under their model?
>
> **A3**. We present a brief discussion about the rank of matrix $W_V$ under some conditions as follows:
>
> Suppose that $SNR \rightarrow \infty$, then we have $\Vert \mu \Vert / \Vert \xi \Vert \rightarrow \infty$. Recall Equation (9) (Line 583), we have:
>
> $\lim \ \nabla_{W_V}L_S(\theta) = ( a \mu_+ + b \mu_- ) w_O^\top$
>
> where a and b are linear combination coefficients for $\mu_+$ and $\mu_- $. Note that $\mu_+$, $\mu_- $ and $w_O$ are vectors (rank = 1), so we have:
>
> $R(\lim \ \nabla_{W_V}L_S(\theta)) = 1$
>
> Considering that the weights of model are usually initialized to be very small, matrix $W_V$ tends to decrease its rank during training because it changes towards its gradient direction $\nabla_{W_V}L_S(\theta)$. The analysis of QK is also similar.
>
> As for a more rigorous proof, we will consider developing new theoretical techniques to prove it.
>
> ---
>
> **Q4**. If an MLP with ReLU activation was used instead of the simple linearly separable task, would the optimization dynamics converge faster or slower?
>
> **A4**. Kou et al. (2023)[2] investigate the benign overfitting phenomenon in two-layer ReLU CNN. Note that their data model only contains one patch of signal and one patch of noise, we can consider their CNN model to be a special MLP, especially in cases of very high or low SNR. Their results show that their model can converge to training loss $\epsilon$ at $t = \eta^{-1} poly(\epsilon^{-1}, d, n, m)$, while the ViT model we consider can converge at $t = \eta^{-1} poly(\epsilon^{-1}, \Vert \mu \Vert, \Vert w_O \Vert)$. Ignoring parameters such as network size, sample size, norm of input and so on, we found that their convergence time is proportional to $\eta^{-1}$ and $\epsilon^{-1}$. So we can think that their convergence speed is of the same order.
>
> ---
>
> **Q5**. Would the SNR requirements be stricter or more relaxed?
>
> **A5**. According to Condition 4.1. in Kou et al. (2023)[2], they require $\mathrm{SNR}^2 \le \tilde{O}(1/N)$ and require $\Vert \mu \Vert_2$ to be large enough, which is stricter than our condition.
>
> ---
>
> Reference
>
> [1] Cao, Y., Chen,Z., Belkin,M., and Gu,Q. Benign overfitting in two-layer convolutional neural networks. In NeurIPS 2022
>
> [2] Kou, Y., Chen, Z., Chen, Y., and Gu, Q. Benign overfitting in two-layer ReLU convolutional neural networks. In ICML 2023

---

> ### Author Response · Authors · 2024-08-13
>
> It will be very encouraging that you can reconsider raising your score if we have addressed all the issues you raised. Otherwise, we are happy with the further discussion.

---

### Official Review · Reviewer_UpG9 · 2024-07-10

**Soundness:** 3
**Presentation:** 2
**Contribution:** 3
**Rating:** 6
**Confidence:** 3

**Summary:**

The paper investigates **the benign overfitting phenomenon in Vision Transformers**. The authors adopt a theoretical framework similar to that proposed by Cao et al. (2022), which use **a data model consisting of label-dependent signal and label-independent noise**, but employ **a two-layer Transformer architecture** instead of a two-layer convolutional neural network. They provide conditions under which benign and harmful overfitting occur.

**Strengths:**

* The paper provides **conditions under which benign and harmful overfitting occur** in a two-layer transformer, with results that are **tight up to a constant**.
* The authors overcome challenges in analyzing the highly complex training dynamics of transformers by introducing **a novel technique called vectorized Q&K and scalarized V**, successfully addressing the learning dynamics. To the best of my knowledge, this is **the first work to successfully address the learning dynamics of even a simple transformer architecture without unrealistic assumptions** (e.g., merging the key-query weights). If all proofs are correct, this represents a significant technical contribution.

**Weaknesses:**

While I believe that the vectorized Q&K and scalarized V techniques are significant contributions, **it is difficult to verify the correctness of the proofs** due to readability issues in both the main text and the appendix. I suggest that the authors **improve the clarity and writing of their technical terms and proofs**.

Minors and Typos
* The notion of $\mu$ in Definition 3.1 seems unnecessary
* Line 9: modal→model
* Line 541: Eexperimantal Rresults → Experimental Results

**Questions:**

* In the data distribution (Definition 3.1), what is **the role of the larger noise $\xi_2$ in the analysis**? What would happen if the data distribution were the same as that considered in Cao et al. (2022), which consists of a single signal patch and a single noise patch?
* In the numerical results section, it would be beneficial to compare the results with those for a two-layer convolutional neural network, as considered in Cao et al. (2022), to emphasize the advantages of the transformer architecture.

Reference

[1] Cao, Y., Chen,Z., Belkin,M., and Gu,Q. Benign over fitting in two-layer convolutional neural networks. In NeurIPS 2022

---

> ### Author Rebuttal · Authors · 2024-08-05
>
> Thanks for your constructive feedback! We address your questions and concerns as follows.
>
> ---
>
> **Weakness**. Readability issues
>
> **A**. We realize that readability is important for readers to understand and further apply our techniques. In order to enhance readability, we have made the following efforts:
>
> - We present a proof sketch in the main text, which contains three main challenges and our solutions, allowing readers to quickly access our techniques.
> - We provide a notation table in page 16, containing the key shorthand notations in this paper.
> - In the appendix, we separate the high-level proof process from the low-level proof process. For example, in appendix F, we present some bounds for $\alpha$, $\beta$ and so on (low-level). In appendix D, we use the bounds in appendix F to complete the proofs for the key steps in the training dynamics (high-level).
>
> We will make more efforts to further improve the readability as follows:
>
> - extend the proof sketch: We consider providing a more comprehensive proof sketch in the appendix to fully describe the key steps in the proof process.
>
> - simplify the notations: The large number of symbols makes it difficult for readers to recall the meaning of a particular symbol. So we will do efforts to make it easier to remember and understand.
> - improve the clarity and writing: We will add more textual descriptions and remarks during the proof process to help readers better understand the proofs.
>
> ---
>
> **Q1**. In the data distribution (Definition 3.1), what is the role of the larger noise $\xi_2$ in the analysis?
>
> **A1**. By Condition 4.1.(10), we ensure that the norm of $\xi_2$ is sufficiently larger than that of other noises. Then under harmful overfitting regime, it becomes easier to prove that $\xi_2$ attract most of the attention and the model learns little signal, thus the test loss will be high.
>
> Without Condition 4.1.(10), it is much more challenging to prove harmful overfitting results. Next we discuss the difficulty we face when $\xi_2$ share the same variance with other noises ($\tilde{\sigma}_p = \sigma_p$)
>
> To prove harmful overfitting, we must prove that the model learn little signals, i.e., signals attract less and less attention and $W_V$ learn little signals.
>
> To prove that the model learn little signals while the training loss converges, we must prove that the model memorizes the noises, i.e., noises attract most of the attention and $W_V$ memorizes the noises well.
>
> But it is difficult to characterize the attention attracted by the noises. For example, consider the scenario at initialization where there are noises $\xi_i$ and $\xi_j$ satisfy: more attention is paid on $\xi_i$ than $\xi_j$ ( attn($\xi_i$) $>$ attn($\xi_j$) ), and $W_V$ memorizes $\xi_j$ better than $\xi_i$ ( denoted by $\rho_j > \rho_i$ ) (under Gaussian initialization, this situation may occur). As we mention in Section 5.2, QK affect V ,and V affect QK. Therefore, the condition $\rho_j > \rho_i$ may lead to an increase in attn($\xi_j$) and a decrease in attn($\xi_i$) in the next training iteration. Meanwhile, the condition attn($\xi_i$) $>$ attn($\xi_j$) may result in $\rho_i$ growing faster than $\rho_j$. Therefore, in the next iteration, the following situations may happen:
>
> - attn($\xi_i$) $>$ attn($\xi_j$),  $\rho_i > \rho_j$
> - attn($\xi_i$) $>$ attn($\xi_j$),  $\rho_i < \rho_j$
> - attn($\xi_i$) $<$ attn($\xi_j$),  $\rho_i > \rho_j$
> - attn($\xi_i$) $<$ attn($\xi_j$),  $\rho_i < \rho_j$
>
> We cannot know which situation will occur, let alone give them precise bounds. New techniques need to be developed to handle this difficulty.
>
> In this paper, we let $\xi_2$ to be stronger than other noises, thus much attention will be paid on $\xi_2$, and $W_V$ will memorize $\xi_2$ well under harmful overfitting regime.
>
> ---
>
> **Q2**. What would happen if the data distribution were the same as that considered in Cao et al. (2022), which consists of a single signal patch and a single noise patch?
>
> **A2**. It is a special case of our Condition (number of input tokens M=2), and the conclusion will remain unchanged.
>
> ---
>
> **Q3**. In the numerical results section, it would be beneficial to compare the results with those for a two-layer convolutional neural network, as considered in Cao et al. (2022), to emphasize the advantages of the transformer architecture.
>
> **A3**. Thank you for your suggestion. We made the comparison in Lines 176 - 178. We will consider adding numerical results to emphasize it.
>
> ---
>
> Reference
>
> [1] Cao, Y., Chen,Z., Belkin,M., and Gu,Q. Benign overfitting in two-layer convolutional neural networks. In NeurIPS 2022

---

> > ### Comment · Reviewer_UpG9 · 2024-08-08
> >
> > Thank you for the authors' response. It adequately addresses my questions. I hope the readability of the overall technical components, including the appendix, will be improved in the next version based on the points discussed in your response.

---

> > > ### Author Response · Authors · 2024-08-08
> > >
> > > Thank you for your response and for acknowledging that our response adequately addressed your questions. We appreciate your constructive comments and suggestions.
> > >
> > > We will consider your suggestion and try to improve the readability of the overall technical components, including the appendix, in the next version of our manuscript. We will ensuring that our paper is as clear and accessible as possible, and we will carefully review and revise the text to enhance its clarity and coherence.
> > >
> > > Once again, thank you for your valuable input. It has been instrumental in helping us improve our work.

---

### Official Review · Reviewer_tKtT · 2024-07-11

**Soundness:** 3
**Presentation:** 3
**Contribution:** 3
**Rating:** 6
**Confidence:** 3

**Summary:**

This study investigates the theoretical aspects of Vision Transformers (ViT) with a focus on their generalization capabilities, particularly under conditions of benign overfitting. Through a detailed analysis of the optimization process involving a self-attention layer and a fully connected layer, optimized using gradient descent on a specific data distribution modal, this work addresses the complexities introduced by softmax functions and the interdependencies of multiple weight configurations in transformer models. By developing novel techniques, the researchers delineate the training dynamics that lead to effective generalization in post-training scenarios. A key contribution is the establishment of a sharp condition based on the signal-to-noise ratio within the data, which predicts whether a small or large test error will occur. These theoretical findings are supported by experimental simulations, enhancing our understanding of transformers' performance in vision tasks.

**Strengths:**

This paper rigorously analyzes the training dynamics of a simplified ViT model. Specifically, its technical contributions related to "Vectorized Q & K and Scalarized V" and "Dealing with the Softmax Function" may have a broader impact on subsequent theoretical research.

**Weaknesses:**

The current experiments only validate the theoretical results on synthetic datasets. It is recommended that the authors consider adding some experiments on real datasets to test the effects, such as experiments on benign overfitting of the ViT model on MNIST and CIFAR10.

**Questions:**

See in Weaknesses.

---

> ### Author Rebuttal · Authors · 2024-08-05
>
> Thanks for your constructive feedback! We address your questions and concerns as follows.
>
> ---
>
> **Q**. The current experiments only validate the theoretical results on synthetic datasets. It is recommended that the authors consider adding some experiments on real datasets to test the effects, such as experiments on benign overfitting of the ViT model on MNIST and CIFAR10.
>
> **A**. We perform some experiments on MNIST dataset. Please find the results in the uploaded PDF file. The experimental result shows a transition between benign and harmful overfitting regimes. The larger the sample size $N$ and signal-to-noise ratio SNR, the better the generalization performance.

---

> ### Author Response · Authors · 2024-08-13
>
> It will be very encouraging that you can reconsider raising your score if we have addressed all the issues you raised. Otherwise, we are happy with the further discussion.

---

### Official Review · Reviewer_GNLt · 2024-07-12

**Soundness:** 3
**Presentation:** 3
**Contribution:** 3
**Rating:** 6
**Confidence:** 4

**Summary:**

The paper studies the benign overfitting phenomenon for a two-layer Transformer in vision. The paper characterizes the optimization of the ViT through three different phases in training dynamics and finds a sharp separation condition of the signal-to-noise ratio to distinguish the benign and harmful overfitting of the ViT.

**Strengths:**

1. The paper gives a sharp transition between benign and harmful overfitting for ViT, which can be verified by a simulation experiment.
2. The paper proposes a novel method of vectorized $QK$ and scalarized $V$ to simplify the study of Transformer.
3. The paper successfully deals with the challenges caused by softmax and multiple weights.

**Weaknesses:**

1. Lack the introduction of the benign overfitting phenomenon in the first section.
2. It is better to clarify some notations like $\Omega(\cdot)$, $\Theta(\cdot)$, $\omega(\cdot)$...
3. Data generation is specified, thus this model may be a little bit limited.
4. A small typo: (Line 149) $\mu||_2^{-2}$ --> $||\mu||_2^{-2}$

**Questions:**

In Theorem 4.1 and Theorem 4.2, the requirement is related to SNR. Why does $\tilde{\sigma}_p$ not occur in the requirement?

**Limitations:**

The authors have stated the limitations.

---

> ### Author Rebuttal · Authors · 2024-08-05
>
> Thanks for your constructive feedback! We address your questions and concerns as follows.
>
> ---
>
> **Q1**. Lack the introduction of the benign overfitting phenomenon in the first section.
>
> **A1**.  In the first section, we first introduce the Transformers and ViT models, and then the empirical and theoretical studies of ViT, and finally the benign overfitting phenomenon, followed by our contributions. The introduction for benign overfitting is in Lines 35 - 38. Benign overfitting is a phenomenon where "**the test error remains small despite overfitting to the training data**". A more detailed introduction is in the first paragraph of the second section. We plan to add more explanations for benign overfitting.
>
> ---
>
> **Q2**. It is better to clarify some notations like $\Omega(\cdot), \Theta(\cdot), \omega(\cdot)$
>
> **A2**. Thank you for pointing it out, we will add a paragraph to explain them.
>
> ---
>
> **Q3**. Data generation is specified, thus this model may be a little bit limited.
>
> **A3**. The data generation model in this paper follows the common assumptions made in previous studies of benign overfitting ( Cao et al. (2022) [1], Kou et al. (2023) [2]). To demonstrate the practicality of our theoretical results in real-world datasets, we perform some experiments on MNIST dataset. The results are detailed in the uploaded PDF file. The experimental result shows a transition between benign and harmful overfitting regimes. The larger the sample size $N$ and signal-to-noise ratio SNR, the better the generalization performance.
>
> ---
>
> **Q4**. In Theorem 4.1 and Theorem 4.2, the requirement is related to SNR. Why does $\tilde{\sigma}_p$ not occur in the requirement?
>
> **A4**. In Condition 4.1 (10), we require that $\tilde{\sigma}_p = C_p \sigma_p$ and $C_p = 5 \sqrt{M}$. We do not make any requirement on $\Vert \mu \Vert $ and $\sigma_p$ in Condition 4.1 so that they are flexible. For example, if we want $\mathrm{SNR} = \Theta(1)$, then we can choose:
> 	  $\Vert \mu \Vert = \Theta(d^{1/2}) , \sigma_p = \Theta(1)$
> or   $\Vert \mu \Vert = \Theta(1) , \sigma_p = \Theta(d^{-1/2})$.
> This setting enable the input to be more general. In real-world dataset, the normalization methods for data may not be the same. Sometimes images are normalized to 0 - 1, while sometimes  they are 0 - 255.
>
> ---
>
> **References**
>
> [1] Cao, Y., Chen,Z., Belkin,M., and Gu,Q. Benign overfitting in two-layer convolutional neural networks. In NeurIPS 2022
>
> [2] Kou, Y., Chen, Z., Chen, Y., and Gu, Q. Benign overfitting in two-layer ReLU convolutional neural networks. In ICML 2023

---

> ### Author Response · Authors · 2024-08-13
>
> It will be very encouraging that you can reconsider raising your score if we have addressed all the issues you raised. Otherwise, we are happy with the further discussion.

---

### Author Rebuttal · Authors · 2024-08-06

We kindly thank all the reviewers for their time and for providing valuable feedback on our work.

To validate the practicality of our theoretical results in real-world datasets, we perform some experiments on MNIST dataset. The experimental result shows a transition between benign and harmful overfitting regimes. The larger the sample size N and signal-to-noise ratio SNR, the better the generalization performance.

For more details, please see the PDF.

---

### Decision · Program_Chairs · 2024-09-25

**Decision:**

Accept (poster)

**Comment:**

The reviewers unanimously agree that this paper should be accepted and presented at neurips, and the AC sees no reason to overturn this recommendation. However, as noted by the reviewer, there are two caveats:
1) Real experimental results are needed to empirically validate the derivations. The authors added MNIST experiments during the rebuttal, which need to be included in the final version of the paper.
2) The proofs are difficult to read, follow, and verify. It sounds like none of the reviewers, nor the AC, was able to fully follow and verify them. Hence, improved clarity is needed. However, we hope that the publication of this paper attracts further scrutiny by the community and leads to follow-up works.